# Modern groundwater reaches deeper depths in heavily pumped aquifer systems

Melissa Thaw[1], Merhawi GebreEgziabher [1], Jobel Y. Villafañe-Pagán[1,2] & Scott Jasechko [1]✉

Deep groundwater is an important source of drinking water, and can be preferable to shallower groundwaters where they are polluted by surface-borne contaminants. Surface-borne contaminants are disproportionately common in 'modern' groundwaters that are made up of precipitation that fell since the ~1950s. Some local-scale studies have suggested that groundwater pumping can draw modern groundwater downward and potentially pollute deep aquifers, but the prevalence of such pumping-induced downwelling at continental scale is not known. Here we analyse thousands of US groundwater tritium measurements to show that modern groundwater tends to reach deeper depths in heavily pumped aquifer systems. These findings imply that groundwater pumping can draw mobile surface-borne pollutants to deeper depths than they would reach in the absence of pumping. We conclude that intensive groundwater pumping can draw recently recharged groundwater deeper into aquifer systems, potentially endangering deep groundwater quality.

Groundwater resources supply drinking water to billions of people[1,2]. However, groundwater supplies are vulnerable to pollution from surface-borne contaminants, which can accompany precipitation as it infiltrates the land surface and percolates down to the water table[3]. Surface-borne contaminants are disproportionately common in groundwater that is made up of relatively recent precipitation[4–10] known as 'modern groundwater'–defined as groundwater comprised of precipitation that fell more recently than the year 1953. Because surface-borne contaminants are disproportionately common in modern groundwater, understanding the processes that control the depth that modern groundwater reaches is important for evaluating water quality risks in shallower versus deeper wells[11,12].

Several local- and regional-scale studies have demonstrated that groundwater pumping can speed up downward flow rates and enable groundwater to reach deep depths while it is still young enough to be considered 'modern'; such pumping-induced downwelling has the potential to also draw shallow pollutants into deep wells used by municipalities and rural communities[13–15]. However, the prevalence of this pumping-induced downwelling of modern groundwater remains poorly understood, as no continent-wide study has tested if modern groundwater reaches deeper depths in places with higher groundwater withdrawal rates.

The objective of this study is to test for statistical relationships between spatial patterns of groundwater withdrawals and spatial patterns of the depth that modern groundwater reaches. To meet our objective, we analyse thousands of US groundwater tritium ($^3$H) measurements (Fig. 1a) to calculate the depth below which modern groundwater is scarce in US aquifer systems (Methods subsection: 'Groundwater tritium data'). Specifically, we calculated the fraction of each well water sample that is comprised of modern groundwater by comparing measured well water $^3$H activities to historical precipitation $^3$H time series[16] (Methods subsection: 'Modern groundwater calculations'). Elevated well water $^3$H activities indicate that modern groundwater is present in a well water sample[17], because most of the precipitation that fell in the US after the year 1953 was artificially enriched in $^3$H by atmospheric thermonuclear testing[16,18]. After completing our $^3$H-based calculations of the proportion of individual well water samples comprised of modern groundwater, we

[1]Bren School of Environmental Science and Management, University of California, Santa Barbara 93106 CA, USA. [2]Department of Geology, University of Puerto Rico, Mayagüez 00682, Puerto Rico. ✉e-mail: jasechko@ucsb.edu

grouped wells located within the same aquifer system (US aquifer geospatial data from ref. 19). These two-dimensional aquifer system areas are underlain by multiple geologic formations of varying

hydrogeologic characteristics (for hydrogeologic cross sections for each study area, see Supplementary Figs. 99–240). Next, for each aquifer system, we calculated the depth below which modern

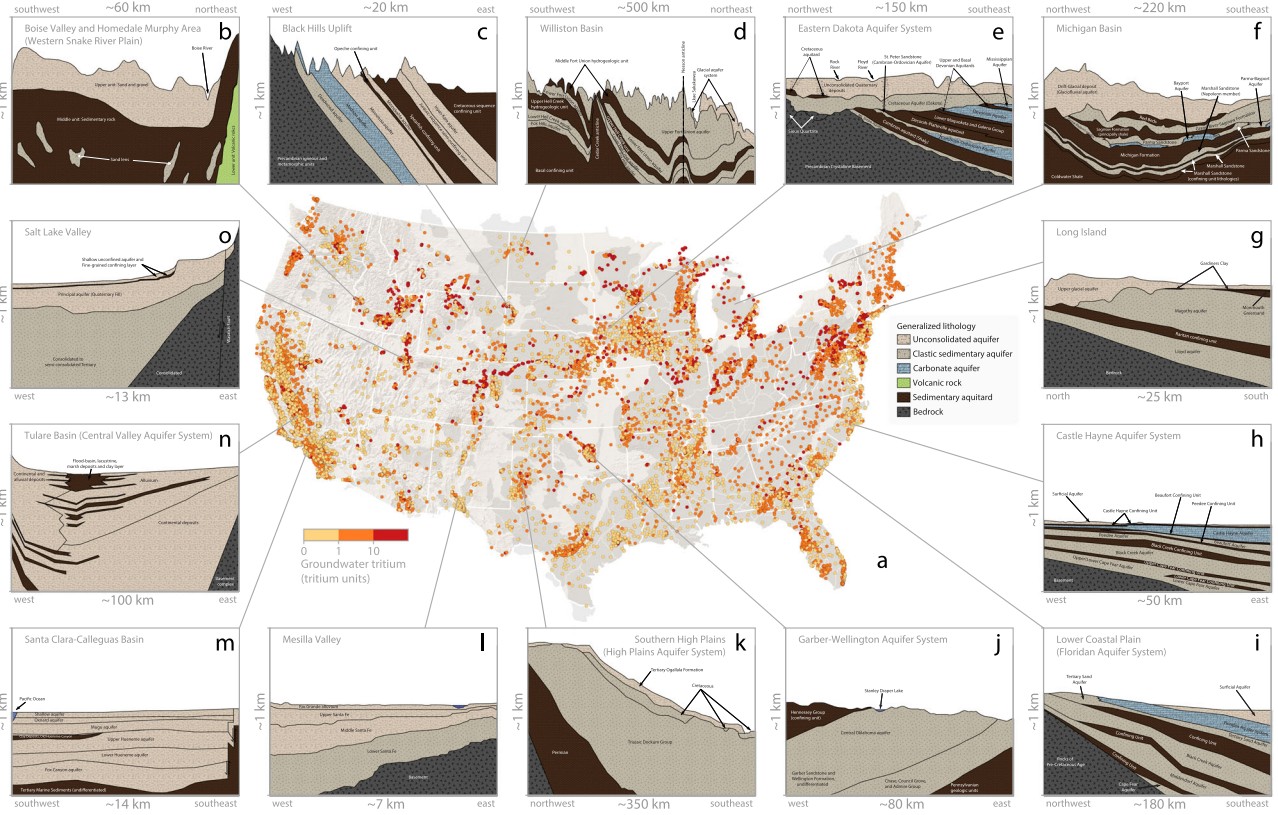

**Fig. 1 | Well water tritium ($^{3}$H) measurements across the contiguous United States. a** Light yellow shades represent lower tritium activities (below 1 tritium unit), orange shades represent mid-range tritium activities (1–10 tritium units) and red shades represent well waters with high tritium activities (exceeding 10 tritium units; 1 tritium unit is ~0.118 Bq/L). Light grey polygons underlying the yellow-red points are aquifer system boundaries published by GebreEgziabher et al. (ref. 19; data available via https://www.hydroshare.org/resource/d2260651b51044d0b5cb2d293d21af08/). **b**–**o** display hydrostratigraphy via cross-sections for $n = 14$ of the $n = 74$ aquifer systems that we studied. The presented cross-sections are based on figures and lithologic descriptions presented for the **b** Boise Valley and Homedale Area within

the broader Snake River Plain in ref. 49, **c** Black Hills Uplift in ref. 50, **d** Williston Basin in ref. 51, **e** Eastern Dakota Aquifer System in ref. 52, **f** Michigan Basin in ref. 53, **g** Long Island in ref. 54, **h** Castle Hayne Aquifer System in ref. 55, **i** Lower Coastal Plain subarea of the broader Floridan Aquifer System in ref. 56, **j** Garber–Wellington Aquifer System in ref. 57, **k** Southern High Plains in ref. 58, **l** Mesilla Valley in ref. 59, **m** Santa Clara-Calleguas Basin in ref. 60, **n** Tulare Basin subarea of the broader California Central Valley Aquifer System in ref. 61, and the **o** Salt Lake Valley in ref. 62. Cross-sections for each of our $n = 74$ study aquifer systems are presented in Supplementary Figs. 99–240 (locations of each of the $n = 74$ hydrogeologic cross-sections we examined are displayed in Supplementary Note 8).

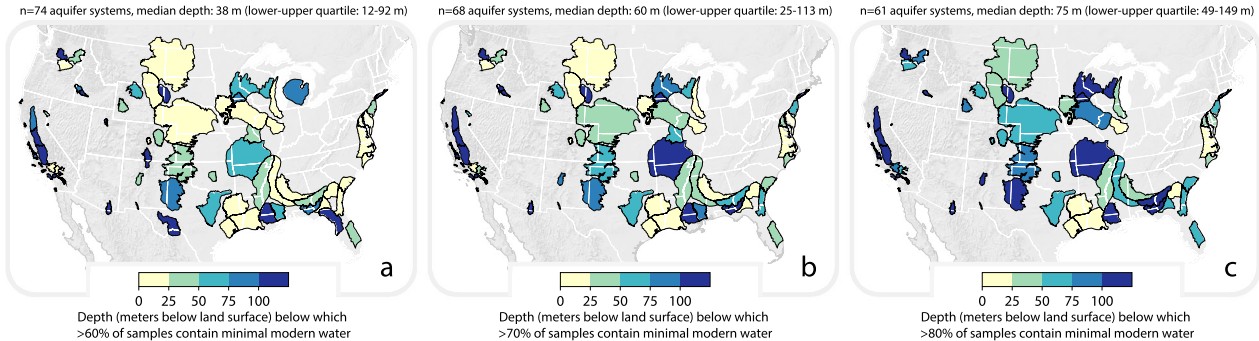

**Fig. 2 | Spatial patterns of the depth below which modern groundwater is scarce among US aquifer systems.** Yellow-blue shades represent the depth below which 60% (**a**), 70% (**b**) or 80% (**c**) of samples contain minimal modern groundwater (defined here as well water samples containing less than 25% modern groundwater). **a** The median depth below which >60% of samples contain minimal modern groundwater among $n = 74$ studied aquifer systems (with sufficient groundwater $^{3}$H data) is 38 m, and the lower–upper quartile range is 12–92 m. **b** The median depth

below which more than 70% of samples contain minimal modern groundwater among $n = 68$ studied aquifer systems is 60 m, and the lower–upper quartile range is 25–113 m. **c** The median depth below which >80% of samples contain minimal modern groundwater among $n = 61$ studied aquifer systems is 75 m, and the lower-upper quartile range is 49–149 m. For a map with labels identifying aquifer systems, see Supplementary Fig. 98.

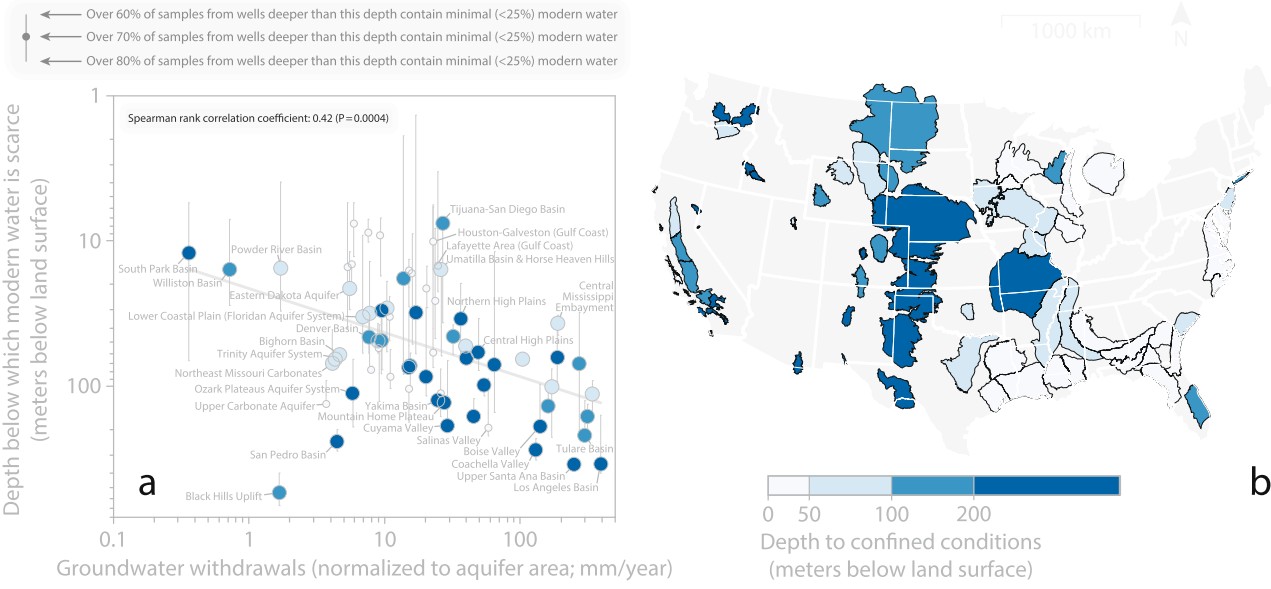

**Fig. 3 | The depth below which modern groundwater is scarce tends to be deeper where annual groundwater withdrawals are high. a** Relationship between estimated annual groundwater withdrawals[21] and the depth below which modern groundwater is scarce. Each point represents one aquifer system. Annual groundwater withdrawals were normalized by aquifer area, such that the units (mm/year) can be interpreted as all withdrawn groundwater during the year 2015 expressed as saturated layer if it were spread evenly across the study area (for study area boundaries, see **b**). The Spearman rank correlation coefficient ($\rho$) of the data presented in panel a is $\rho = 0.42$ (Spearman *P* value <0.001). The *y*-axis values corresponding to each coloured point are the depth below which >70% of samples have minimal (<25%) modern groundwater; vertical error bars extend to the depth below which >60% of samples have minimal (<25%) modern groundwater (shallower depth−i.e., top of grey error bar as displayed in plot) and the depth below which >80% of samples have minimal (<25%) modern groundwater (deeper depth− i.e., bottom of grey error bar as displayed in plot; see legend in upper-left corner of plot). We only plot the *n* = 68 points representing the aquifer systems for which we had sufficient data to determine the depth below which >70% of samples have minimal (<25%) modern groundwater. Points are colour coded by the estimated depth to confined conditions (see legend in **b**). **b** Estimated depth to confined conditions in each of our study aquifer systems. Each polygon on the map represents one study area. Light blue colours represent shallower depths to confined conditions; darker blue shades represent deeper depths to confined conditions. Depths to confined conditions were estimated on the basis of up to three data sources: (i) US Geological Survey defined well conditions (i.e., wells defined as tapping unconfined versus confined conditions by the US Geological Survey), (ii) digitization and evaluation of hydrogeologic cross sections derived from local-scale reports, and (iii) quotes within local-scale reports pertaining to the prevalence of confined conditions (for details see Supplementary Note 3).

groundwater is scarce by calculating the shallowest depth below which most samples (60%, 70% or 80%; Fig. 2) collected from deeper wells contain 'minimal modern groundwater' ('minimal modern groundwater' defined as well water samples containing <25% modern groundwater; Methods subsection: 'Calculating the depth below which modern groundwaters are scarce'). We excluded *n* = 17 aquifer systems where a visual inspection of modern groundwater variations with depth suggests that our approach did not adequately capture the depth below which modern groundwater becomes scarce (Supplementary Note 1, Supplementary Figs. 1–91, and Supplementary Table 1). Last, we estimated groundwater withdrawal rates within the boundaries of each study aquifer system by downscaling county-scale groundwater withdrawal data provided by the US Geological Survey[20], and tested for spatial correspondence between groundwater withdrawals and the depth below which modern groundwater becomes scarce via rank regression (Methods subsection: 'Geospatial analyses of potential explanatory variables'; see Supplementary Note 2 for further details pertaining to our calculations of annual groundwater withdrawals).

## Results and discussion
### Modern groundwater common at shallow depths
We identified *n* = 74 aquifer systems with sufficient well water ³H data to quantify the depth below which most (60%, 70% or 80%; Fig. 2a–c) groundwater samples contain minimal modern groundwater. Among our study aquifers, the median depths below which most groundwater samples contain minimal modern groundwater range from 38 to 75 m (the range of median values derives from our three different quantitative definitions of 'most': 60%, 70%, or 80%).

The depth below which modern groundwater is scarce is relatively shallow (less than 50 m) in portions of the Gulf Coast Regional Aquifer System (Houston-Galveston and Lafayette sub-areas) and in the Northern Great Plains Aquifer System (Williston Basin and Powder River Basin sub-areas; Fig. 2). By contrast, the depth below which modern groundwater is scarce is relatively deep (greater than 100 m) in Long Island, the Los Angeles Basin, California's Central Valley (San Joaquin and Tulare sub-areas), and alluvial basins in Arizona (e.g., San Pedro Basin) and California (e.g., Coachella Valley; for individual plots for each aquifer system see Supplementary Figs. 1–91; for a map with labelled aquifer system titles, see Supplementary Fig. 98).

The observation that modern groundwater tends to be most common at shallow depths has important ramifications for well water quality, as surface-borne contaminants are disproportionately common in modern groundwaters[4–10]. Critically, from a water quality risk perspective, we note that most US drinking water wells are perforated at relatively shallow depths where modern groundwater is most common; 55% of domestic water wells are shallower than 50 m, and 84% are shallower than 100 m (ref. 21); however, we stress that most drinking water wells are domestic water supply wells rather than public water supply wells, and that the latter tend to be deeper than the former. Our finding that most domestic water wells are perforated at relatively shallow depths−where our ³H data suggest modern groundwaters are most common−implies that the water pumped by many domestic wells is dominated by modern groundwater, which is disproportionately likely to contain surface-borne contaminants.

Drilling domestic water wells to deeper depths may reduce the likelihood of well water contamination events in some areas[11,12,22,23]. However, drilling deeper wells to avoid shallow contaminated aquifers

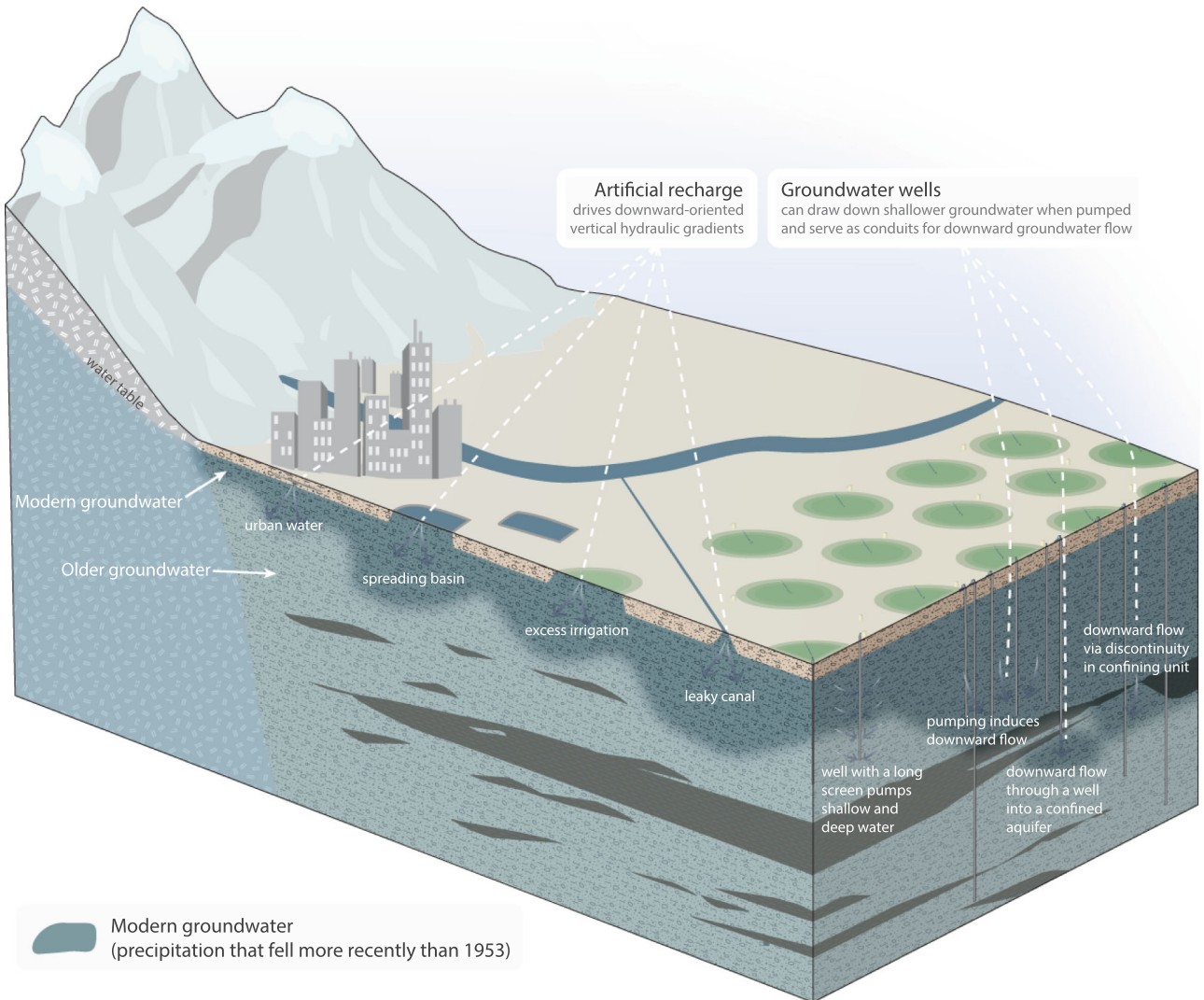

**Fig. 4 | Schematic of some of the processes that may influence vertical distributions of modern groundwater.** Artificial groundwater recharge (left side of schematic) can increase the magnitude of downward-oriented vertical hydraulic gradients and potentially drive modern groundwater to deeper depths. Artificial recharge can derive from urban waters (e.g., Tucson Basin[63]), spreading basins associated with managed aquifer recharge projects (e.g., Upper Santa Ana Basin[64]), excess irrigation waters (e.g., San Joaquin Basin[65]), or leaky surface water conveyance infrastructure such as canals (e.g., Utah Lake Valley[66]). The construction of groundwater wells and their use via pumping (right side of schematic) may help modern waters enter deeper wells. For example, deep wells that have long perforated intervals may draw both shallow (disproportionately modern) and deep

(disproportionately pre-modern) groundwater (e.g., Tulare Basin[67]). Further, pumping from wells can alter vertical hydraulic gradients, potentially drawing shallow modern groundwater to deeper depths (e.g., Salt Lake Valley[68]). The modern groundwater may move downward to deeper depths in the aquifer system via natural pore spaces (e.g., discontinuities in aquitards; e.g., Eastern Mississippi Embayment[34]) or via conduits created by constructed wells themselves (e.g., Northern High Plains[69]). For further discussion and schematics of modern groundwater distributions, see refs. 48, 70, 71. For a review of some of the studies that have posited one or more of these mechanisms as potential explanations for the distribution of modern groundwater, see Supplementary Note 6. For a more expansive version of this figure, see Supplementary Fig. 243.

may be a stopgap[21], if pumping from nearby municipal or irrigation wells draws modern groundwater downward and jeopardizes deep groundwater quality. Nevertheless, the prevalence of pumping-induced downwelling at continental scale is not known. Therefore, we calculated correlations between groundwater withdrawals and the depth below which modern groundwater is scarce (see next section entitled: 'Deeper modern water where groundwater withdrawal rates are high'; see Methods subsection: 'Geospatial analyses of potential explanatory variables').

### Deeper modern water where groundwater withdrawal rates are high

We find a significant (Spearman $P$ value < 0.01) positive correlation between annual groundwater withdrawals and the depth below which modern groundwater is scarce (Fig. 3a). Spearman rank correlation

coefficients ($\rho$) range from $\rho = 0.39$ to $\rho = 0.42$ (all statistically significant at $P$ value < 0.01; the range of $\rho$ values derives from three correlation coefficients, each based on the depth below which 60%, 70% or 80% of samples contain <25% modern groundwater; Supplementary Table 78). Our finding suggests that modern groundwater tends to reach deeper depths in aquifer systems that are heavily pumped.

It is plausible that shallow low-permeability geologic formations could limit the depth below which modern groundwater is scarce and influence the effects of pumping on vertical variability in groundwater age[24]. We estimated the depth to confined conditions for each of our $n = 74$ study areas by analysing (i) vertical variations in the proportion of wells defined as tapping confined aquifers by the US Geological Survey, (ii) hydrogeologic cross sections derived from local-scale studies, and (iii) statements pertaining to confined conditions from local-

scale studies (see Fig. 3b; for detailed approach for estimating depth to confined conditions for each of our $n = 74$ study aquifers see Supplementary Notes 3.1–3.74). We show that the depth below which modern groundwater is scarce tends to be deeper in aquifer systems characterized by thicker unconfined zones (i.e., aquifers with a relatively high (i.e., deep) depth to confined conditions; see Supplementary Note 4).

Therefore, the significant positive correlation between annual groundwater withdrawals and the depth below which modern groundwater is scarce (Fig. 3a) could arise spuriously if groundwater pumping tends to be higher in aquifer systems characterized by thick unconfined zones. To account for potential interrelationships between groundwater withdrawals and the depth to confined conditions, we completed multiple regression of the rank transforms of each explanatory variable[25] (Supplementary Notes 4 and 5). The resulting partial regression coefficients ($\beta$)—that describe the statistical relationship between groundwater withdrawals and the depth below which modern groundwater is scarce—remain positive and significant ($\beta$ values range 0.29 to 0.34; all significant at Spearman $P$ value < 0.05; see Supplementary Table 79). This analysis suggests that our finding that modern groundwater tends to reach deeper depths in aquifer systems that are heavily pumped holds even after accounting for differences in the depth to confined conditions among our study aquifers.

Our pan-US statistical analyses are consistent with local-scale research in California's Central Valley[26,27], where groundwater pumping draws young and shallow groundwater deeper into the aquifer system; our results suggest that pumping-induced downwelling is not unique to California's Central Valley and is likely occurring in other heavily pumped US aquifer systems. Though we find that modern groundwater tends to reach deeper depths in heavily pumped aquifer systems (Fig. 3a), we emphasize the moderate strength of the correlation and the high proportion of unexplained variance (see substantial scatter in points in Fig. 3a). We also emphasize that the groundwater withdrawal data that we analyse[21] are highly uncertain; further, the county-scale at which these data are available limit their local relevance and complicate geospatial analyses (see Methods subsection entitled: 'Limitations'). There is considerable room for improvement in US groundwater withdrawal data[28]. Should better groundwater withdrawal data become available, these data would enable a better assessment of the statistical relationship between groundwater withdrawals and the depths that modern groundwaters reach.

### Potential explanations for deep modern groundwater
While our tritium data set cannot identify the specific mechanisms that transmit modern groundwater to deeper depths where groundwater withdrawal rates are high, there are a number of potential mechanisms that may help explain our main finding—that modern groundwater tends to reach deeper depths in aquifer systems that are heavily pumped (Fig. 4 and Supplementary Note 6).

First, pumping from deep aquifers can alter vertical hydraulic gradients and speed up downward flows of shallow groundwaters[13,29–32]; the existence of preferential flow pathways through permeable geologic structures (e.g., discontinuities in aquitards at faults or permeable intercalations[33]) may further enhance pumping-induced downward transport of modern groundwater[34]. Second, the magnitude of downward-oriented vertical hydraulic gradients—which drive modern groundwater to deep depths—may be increased by artificial groundwater recharge associated with land use and water management[26] (e.g., infiltration of excess irrigation waters[35,36], leakage from urban water infrastructure[37] or canals[38]). Spreading basins and intentional flooding have been applied via 'managed aquifer recharge' projects for decades in several heavily pumped aquifer systems in California and Arizona[39,40], potentially increasing downward groundwater flow rates in these areas. Recharge induced by such processes can sustain high-

magnitude and downward-oriented vertical hydraulic gradients and may drive modern groundwaters to deeper depths. Third, modern groundwaters may be present in some deep aquifers because they were intentionally injected via aquifer storage and recovery projects, but these projects are not common in many of our study areas[41]; this mechanism is speculative but plausible. Fourth, modern groundwater may be transmitted relatively rapidly to deep depths in heavily pumped aquifer systems that also host poorly sealed wells, which can create conduits that enable rapid movement of shallow modern groundwater to deep aquifers[33,42–44].

To better understand the potential for confining units to limit the depth that modern groundwater reaches, we examined the prevalence of modern groundwater in $n = 1831$ groundwater samples where the well has been defined by the US Geological Survey as tapping a confined aquifer (well metadata field aqfr_type_cd specifies 'Confined single aquifer' or 'Confined multiple aquifers'). We find that about one-third of groundwater samples collected from a well that is perforated in a confined aquifer ($n = 592$) contain more than 5% modern water, highlighting that modern water can enter wells screened in confined aquifers (Supplementary Note 7 and Supplementary Fig. 244).

Differences among our study aquifers in pumping well depths, well integrity, land use activities and managed aquifer recharge practices likely contribute to the observed variability in the depth below which modern groundwater becomes scarce in our study areas. Identifying locally relevant mechanisms that may rapidly transmit shallow groundwater to deep depths is important to create strategies to protect deep groundwater quality.

### Pumping-induced downwelling may impact groundwater quality
Although our tritium data cannot identify specific transport mechanisms for each of the dozens of aquifer systems we studied, our analyses demonstrate a moderately strong statistical relationship that suggests pumping may lead modern groundwaters to reach deeper depths than they would flow to naturally (Fig. 3a).

Where modern groundwater is contaminated, pumping-induced downwelling of these groundwaters can threaten deep groundwater quality; however, we stress that many contaminants flow at considerably slower rates than the groundwater itself due to, for example, retardation via adsorption. The downwelling of modern groundwater may also have indirect impacts on groundwater quality. For example, in the Red River Delta (Vietnam), intensive pumping of deep aquifers has likely increased aqueous arsenic concentrations both directly via the downwelling of shallow arsenic-rich groundwaters to deeper depths[30], and also indirectly as these downward-flowing groundwaters contribute arsenic-mobilizing solutes[45]. Because deep aquifers tend to require millennia to flush, contamination of deep groundwaters can be especially challenging to remedy, and some groundwater remediation technologies are not well suited for deep groundwater.

Deep groundwater is a globally important water supply, and its value is expected to grow where shallower groundwater stores and qualities are declining and deteriorating. We demonstrate that modern groundwater tends to reach deeper depths in heavily pumped aquifer systems, signalling that pumping can rearrange groundwater flow-paths and impact deep groundwater quality.

## Methods
### Groundwater tritium data
We downloaded US well water tritium measurements from the water quality portal (https://www.waterqualitydata.us/portal accessed April 20, 2021). We excluded tritium measurements that were below analytical detection limits if the reported detection limit exceeded 0.8 tritium units (where 1 tritium unit equals 0.118 Bq/L). We excluded all measurements of media that were not categorized as 'Groundwater' (field code: 'ActivityMediaSubdivisionName'). We deleted one record

with a code reading 'Systematic Contamination'. We converted negative values to zeros. We excluded one well water measurement with an unlikely 'ActivityStartDate' value of September 28, 1900. If more than one tritium measurement was available for a single well, we analysed only the most recent record (i.e., the measurement with the most recent date) and excluded the other measurements, to ensure each well was only counted once in our analysis.

We excluded all records that did not report a numeric well depth value (i.e., both of the following fields were blank: 'WellDepthMeasure/ MeasureValue' and 'WellHoleDepthMeasure/MeasureValue') or recorded a value of zero for the well depth. If a non-zero 'WellDepthMeasure/ MeasureValue' was available, we prioritized analysis of this well depth value; otherwise, we included the value of 'WellHoleDepthMeasure/ MeasureValue' as the well depth. We emphasize that both values (i.e., 'WellDepthMeasure/MeasureValue' and 'WellHoleDepthMeasure/MeasureValue') are likely to be approximations because of the inherent uncertainties associated with well construction. We also highlight that our data set does not provide information about perforated intervals for all of our wells, requiring us to analyse total well depths rather than screen depth intervals and introducing uncertainty into our analyses (see (ii) in the Limitations section below).

## Modern groundwater calculations

We downloaded geospatial time series of estimated precipitation tritium for the contiguous US (from https://www.sciencebase. gov/catalog/item/5af49307e4b0da30c1b44e10; see ref. [16]). For each groundwater well for which we have tritium data, we identified the nearest precipitation tritium time series grid (in most cases, the well location lies within a precipitation tritium time series grid cell). Next, we decay-corrected all values in the precipitation tritium time series to determine the range of possible 'net present' tritium activities relevant to a given groundwater tritium measurement (following method by ref. [46]). To account for the strong likelihood that at least some hydrodynamic dispersion takes place as groundwater flows within aquifer systems, we smoothed these decay-corrected precipitation tritium time series by a five-year running average (following methods by ref. [47]). We then calculated the maximum and minimum decay-corrected post-1953 precipitation tritium values from these running averages; hereafter these values are used as high and low values for the term $^{3}H_{post-1953}$. We then straightforwardly calculated the fraction of each groundwater sample comprised of 'modern groundwater' derived from precipitation that fell since the year 1953 ($F_{post-1953}$) following ref. [46]:

$$F_{post-1953} = \frac{^{3}H_{sample} - {^{3}H_{pre-1953}}}{^{3}H_{post-1953} - {^{3}H_{pre-1953}}} \qquad (1)$$

$^{3}H_{sample}$ represents the measured tritium activity in the well water. $^{3}H_{post-1953}$ and $^{3}H_{pre-1953}$ represent the estimated local precipitation tritium activities[16] after correcting for radioactive decay to the time of sampling. $^{3}H_{post-1953}$ represents the decay-corrected precipitation tritium for years between 1953 and the date the sample was analysed (applying the aforementioned five-year running average). $^{3}H_{pre-1953}$ represents the decay-corrected tritium activity for precipitation that fell prior to 1953 (assumed to be zero, as most of groundwater $^{3}H$ samples in our data set were analysed after multiple tritium-half-lives have elapsed since 1953, and pre-1953 precipitation tritium activities in the US were likely less than 10 tritium units[48]).

## Calculating the depth below which modern groundwaters are scarce

We then created a binary categorization for each sample, defining each sample as either (i) containing minimal modern groundwater, defined as samples with a maximum $F_{post-1953}$ value of

<0.25 (i.e., <25% of the sample is comprised of modern water), or (ii) samples with a maximum $F_{post-1953}$ value exceeding 0.25. The maximum $F_{post-1953}$ values were determined for each sample using the minimum $^{3}H_{post-1953}$ values in Eq. 1 (i.e., entering the minimum $^{3}H_{post-1953}$ value into Eq. 1 yields the maximum $F_{post-1953}$ value for a given groundwater sample).

First, we grouped all groundwater tritium samples by the aquifer system that the well lies within. To avoid analysing the most data-sparse aquifers, we only analysed aquifer systems with (a) at least $n = 10$ tritium measurements, and (b) at least a 100 m vertical offset between the total depths of the shallowest well and the deepest well for which we have groundwater tritium measurements.

Second, for each well within each study aquifer, we calculated the proportion of all samples that derive from wells that are deeper than a given depth that contain minimal modern water (i.e., categorization (i) above). We did not calculate these proportions if fewer than five wells were deeper than the depth of interest (i.e., we require a minimum of five data points to calculate a value of 'the proportion of wells with deeper depths that contain minimal (i.e., maximum $F_{post-1953}$ is <25%) modern water'). For each aquifer system, we determined the depth at which 60, 70, or 80% of samples derived from wells deeper than this depth contain <25% modern water (these depths presented in Fig. 2).

Third, to further scrutinize our results, we visually inspected individual plots of modern groundwater prevalence with depth for each of our study aquifers (Supplementary Figs. 1–91). We identified $n = 17$ aquifer systems where our approach (i.e., calculating the depth below which 60, 70 or 80% of samples contain <25% modern groundwater) provided an imperfect estimate of the depth below which modern groundwater becomes scarce due (a) a lack of groundwater tritium data at depth, or (b) a lack of a consistent decline in modern groundwater fractions with depth (Supplementary Table 1 and Supplementary Figs. 4, 6, 9, 12, 16, 19, 33, 37, 44, 52, 54, 59, 71, 79, 82, 87, 88). We excluded these $n = 17$ aquifer systems from our analyses.

## Geospatial analyses of potential explanatory variables

For each study aquifer system with sufficient groundwater tritium data for analyses, we compared our estimates of 'the depth below which modern groundwater is scarce' (i.e., values in Fig. 2) to two potential explanatory variables: (a) annual groundwater withdrawals (from: https://www.sciencebase.gov/catalog/item/get/5af3311be4b0da30c1b 245d8), and (b) the depth to confined conditions.

(a)   To estimate groundwater withdrawals within each of our study aquifers (i.e., potential explanatory variable (a)), we downscaled county-scale groundwater withdrawal data by analysing irrigated area and population density data within individual county areas, thereby providing greater statistical weight to portions of counties where groundwater withdrawal rates are more likely to be relatively high. For further details see Supplementary Note 2.

(b)   To estimate the depth to confined conditions within each of our study aquifers (i.e., potential explanatory variable (b)), we analysed (i) vertical variations in the proportion of wells defined as tapping confined aquifers by the USGS (depth to confined conditions estimated as the shallowest depth where both (a) the fraction of wells that tap confined conditions exceeds 80%, and (b) more than 80% of wells at deeper depths are classified as tapping confined conditions), (ii) hydrogeologic cross sections derived from local-scale studies, and (iii) statements pertaining to confined conditions from local-scale studies. For further details see Supplementary Note 3.

We then calculated rank correlation coefficients describing variability in our calculated values for the depth below which modern

groundwater is scarce and these two explanatory variables (see Supplementary Note 4). Our calculations include independent correlations for each potential explanatory variable (Supplementary Table 78), and also a multiple regression of the rank transforms to account for potential interrelationships among the explanatory variables themselves[25] (Supplementary Table 79).

## Limitations

Our analyses have a number of limitations, five of which are described here:

(i)   First, the groundwater withdrawal data[20] we analysed are highly uncertain and available only at county-scale. The substantial uncertainties in groundwater withdrawal data arise in part due to the lack of widespread groundwater withdrawal metering data, which limits our confidence in these withdrawal estimates (i.e., x-axis values in Fig. 3a). Further, the county-scale at which these groundwater withdrawal data are reported[20] complicates our interpretation of groundwater withdrawal data and required us to downscale groundwater withdrawal data to finer spatial resolution, introducing uncertainty in the process. We emphasize that the groundwater withdrawal estimates reported in Fig. 3a and our correlation coefficients are uncertain, and that the correlation coefficients we present would differ if better groundwater withdrawal data were to become available.

(ii)  Second, our evaluation of the depths below which modern groundwater is scarce is limited by the lack of perforated interval data for some of the wells where groundwater tritium measurements have been made. Our calculations are thus necessarily limited to total well depth data, which are, if anything, an overestimate of the depth of the groundwater sampled by the well. Wells that are deeper than the total depths that modern groundwater penetrates may pump modern groundwater because some of these wells will be perforated at shallower depths where modern groundwater is more abundant, potentially leading to overestimations of the depth below which modern groundwaters penetrate in some wells with extensive perforated intervals.

(iii) Third, we cannot evaluate the depths at which groundwater withdrawals take place, as the data we analyse[20] are two-dimensional data products that do not contain information about the depths at which pumping rates are higher versus lower. However, all of our study aquifers have at least some recorded wells[21] with depths that exceed the depth below which modern groundwater is scarce, implying a potential for groundwater withdrawals to play a role in determining the depth at which modern groundwater becomes scarce.

(iv)  Fourth, we analyse groundwater withdrawals for the year 2015 but acknowledge that groundwater pumping that took place prior to the year 2015 has also likely contributed to the depth distribution of modern groundwater.

(v)   Fifth, our estimate of a single value for the depth to confined conditions for a given aquifer system oversimplifies real world heterogeneous conditions. The true depth to confined conditions likely varies, possibly substantially, within each of the $n = 74$ aquifer systems that we study. We stress that the depths to confined conditions presented in Fig. 3b and Supplementary Note 3 are approximations.

## Data availability

Aquifer system boundaries from ref. 19 are available to download via CUAHSI HydroShare: https://www.hydroshare.org/resource/d2260651b51044d0b5cb2d293d21af08/. Groundwater withdrawal estimates (county-scale) are available via https://pubs.er.usgs.gov/publication/cir1441. Well water tritium data are available via https://www.waterqualitydata.us/portal.

## Code availability

The analyses presented here do not depend on specific code. Our approach can be reproduced following the procedures described in the 'Methods'.

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

## Acknowledgements

This material is based upon work supported by the National Science Foundation under Grant No. EAR-2048227 (to S.J.). This research was supported by funding from the Zegar Family Foundation (to S.J.). This material is based upon work supported by the U.S. Geological Survey (USGS) through the California Institute for Water Resources (CIWR) under Grant/Cooperative Agreement No. G21AP10611-00 (to S.J.). The views and conclusions contained in this document are those of the authors and should not be interpreted as representing the opinions or policies of USGS/CIWR. Mention of trade names or commercial products does not constitute their endorsement by USGS/CIWR. The authors acknowledge the Jack and Laura Dangermond Preserve (https://doi.org/10.25497/D7159W), the Point Conception Institute, and the Nature Conservancy for their support of this research.

## Author contributions

M.T. and S.J. contributed equally to the manuscript. M.T. and S.J. co-generated hypotheses, co-developed methods, and co-wrote the manuscript. M.T. and S.J. calculated the depth below which modern groundwater is scarce. M.G. digitized hydrogeologic cross sections and completed the literature review to categorize geologic formations (e.g., as aquifers or as aquitards). J.Y.V-P. georeferenced maps from reviewed literature to approximate the location of each hydrogeologic cross section. S.J. estimated the depth to confined conditions for each study area.

## Competing interests

The authors declare no competing interests.
