## [Peer Review File · Nature Communications]

Reviewer comments, first round review

Reviewer #1 (Remarks to the Author):

Many parts of the US rely on groundwater as a source of clean drinking water and as a source of irrigation water. Aquifers that are heavily pumped and where recharge is enhanced by irrigation or managed aquifer recharge can increase the vertical movement of groundwater than would exist if these human activities were absent. However, it is a difficult task to show a relationship between pumping and the penetration depth of modern water.

The authors compiled an extensive digital dataset of aquifer boundaries and computed the fractions of modern water for wells within each aquifer. They show the depth where modern water is scarce based on the depths where 60%, 70%, and 80% of wells have small fractions of modern water. They use groundwater withdrawal data to show that pumping has marginal relationship with the depth of scarce modern.

Intuitively, it reasonable to expect that the vertical movement of groundwater is enhanced by pumping, however, the data and analysis is not concretely supportive of this hypothesis. I think part of this stems from the fact that some aquifers are unconfined while others are largely confined, with smaller areas of aquifer that are unconfined (Cambrian-Ordovician, Mississippi-Embayment for example). These distinctions are important because unconfined aquifers will draw water downward due to gravity drainage whereas confined aquifers are more likely to draw water locally. Consequently, a heavily pumped unconfined aquifer may increase the vertical penetration of modern water whereas a heavily pumped confined aquifer could actually increase the age of old groundwater in a well (see Zinn and Konikow, 2007; doi:10.1029/2006WR004865). As a result, the effect of pumping on penetration depth of modern water might be easier to see in unconfined aquifers rather than by lumping them all together or differentiating them by lithology.

For the statistical relationship to be valid, there must also be some statistical relationship within some groups of the ensemble. The graph is not very convincing. Maybe the relationship is just a result of grouping many aquifers with different lithologies/confinement/thicknesses? As it stands now, the statistical analysis in lines 144-163 is not that supportive of the main result and is a distraction. Also, it would be expected that the alluvial basins would be the most supportive of the author's conclusion, but the graph alone is not enough to support that inference and the author's do not include the statistical relationship for this set of aquifers in the text and not the table. Unfortunately, the statistical relationship between pumping and depth is not strong either. But I think it might useful to discuss the variations and reasons for deeper penetration of modern water within this aquifer type, rather than the statistical analysis in lines 144-163. It would also motivate a discussion of why the relationship is lacking in other aquifer types, perhaps due to confinement?

The other aspect that is difficult to assess is the effect of the assigned groundwater withdrawals on the correlation with penetration depth. The step to compute the withdrawal weight isn't clear and the area-weighting could have a large influence.

The other explanatory variables were not related to depth of scarce modern water, so either explain why they were not related or omit.

LINE COMMENTS:

Lines 45-47: I think you mean that groundwater pumping can draw groundwater to deeper depths more rapidly than it would naturally. As it is written now, it implies the modern water wouldn't ever reach that depth but that is not true, it just would take longer in the absence of pumping.

Lines 60-62: How do you differentiate aquifers when wells are located in areas where two aquifers are present. For example, sites in Iowa could be screened in the Glacial aquifer system or the Cambrian-Ordovician aquifer system.

Lines 89-90: US drinking water wells could mean public supply wells or domestic wells. It looks like you're referring to only domestic well uses (~44 million people) and not public supply well (~110 million people). These two types of wells have very different well construction characteristics and pumping characteristics. Public supply wells are typically screened deeper in the system than domestic wells, yet this paragraph seems to imply that most people are drinking shallow, modern water and that is not true.

Lines 96-99: Pumping from domestic wells isn't really the problem because they pump infrequently and at lower volumes than public wells. Pumping from public supply wells, however, can perturb the systems substantially, particularly when they're concentrated in urban areas, like the SJV for example.

Lines 144-149: Be consistent when talking about variables in a statistical context. I assume you mean that groundwater pumping = groundwater withdrawals, which was defined on line 131 and used in line 135.

Lines 144-149: I do not understand this. Groundwater pumping and depth below which modern groundwater is scarce was positive and significant in previous paragraph, so why prove it again? The first sentence seems to suggest you are going to understand how groundwater pumping and the other explanatory variable are related, which may help explain why the variables aren't correlated with depth. But then the other variables are not discussed.

Lines 151-163: This does not add anything to the discussion. It might be best to exclude those aquifers from the beginning.

Lines 165-168: Two out of the three correlations have p-values greater than 0.05 and the rho range from 0.34 to 0.38, so these are not strong correlations.

Lines 183-194: Please define the units of groundwater withdrawals. I don't think is millimeters per year unless you took the total and divided it by the aquifer boundaries, in which case please state that in the caption or maybe label it millimeters per square meter of aquifer per year.

Lines 207-209: This should be the last explanation not the first, particularly because it is speculative, although I agree it can be a factor. The part that is missing from this analysis, is the recharge component. Pumping alone will draw modern water downward but it becomes diminished as the head gradient decreases over time. But in places where irrigation is applied, like the central valley or high plains, or where recharge managed, like the LA Basin, the gradients can be sustained and provide an additional driving force for the deeper penetration of modern groundwater. This needs to be spelled out in the paper and it should be the second explanation for deep penetration of modern water, not the third. The first should be pumping. I think it short-circuit effects are important and are common but it needs to more study to quantify it's impact more broadly.

Lines 219-222: The fourth seems speculative but plausible.

Lines 303-306: I thought there were two $3H_{\text{post-1953}}$ cut off values used? Which one was used as the basis for computing $F_{\text{post-1953}}$? Does the maximum $F_{\text{post-1953}}$ correspond to the minimum $3H_{\text{post-1953}}$ threshold?

Line 333: You mean groundwater withdrawals or is there another groundwater pumping dataset?

MINOR COMMENTS

Supplementary notes are not referenced in order.

SUPPLEMENTAL INFORMATION COMMENTS

Give the development of a new aquifer system database, it would seem appropriate to have this published on its own and receive peer review prior to it's use. There are many places where one

aquifer overlies another. For example, the Surficial aquifer overlies the Floridan in parts of Florida. How do you assign a well to the aquifer? Since this dataset is composed largely of USGS data, did you use the aquifer codes associated with each well to help differentiate aquifer assignments to wells?

Did you really map hundreds of aquifer systems or did you compile, digitize, georeference numerous mapped aquifers?

Supplementary Table 7. Are these meters?

Reviewer #2 (Remarks to the Author):

This study utilized existing, publicly-available data to investigate the potential controls on modern groundwater depths within numerous aquifer systems in the USA. The authors find a significant dependence on groundwater pumping (i.e., the penetration depth of modern groundwater is greater in aquifer systems with higher overall pumping rates), as revealed by tritium concentrations in well water samples. The geospatial, mapping, and statistical analyses are impressive. The results of the study do not directly advance hydrologic understanding because the relationship between pumping and modern groundwater depth is not mechanistically explained for the various aquifer settings considered. However, this manuscript offers a valuable compilation of existing data, with novel geospatial analysis, which will be useful to the water resources research community. I recommend that the paper be published after some moderate revision. Specific comments and questions are listed below.

(1) As the authors are likely aware, aquifer properties (transmissivity, storage coefficients) and aquifer structure (e.g., heterogeneity, presence of confining units) are important controls on the penetration depth of modern groundwater. Although these factors were not included as explanatory variables, the grouping of aquifers by geologic setting (Figure 3) is a useful exercise that partially addresses this. However, I think the manuscript would benefit from some additional consideration of these factors, at least with added discussion and/or description of limitations. For example, note that some of the aquifer systems are multi-aquifer systems with regionally extensive confining units (e.g., Mississippi Embayment, Trinity Aquifer System, Denver Basin, and others). The presence of low-K confining units that limit vertical hydrologic connectivity would seem to be an important control on the depth of modern groundwater.

(2) line 131: It seems that explanatory variable (iii) would have different meaning for the different types of aquifer systems considered in this study. For bedrock aquifers, this is more clearly interpretable as overlying regolith and sediment. For alluvial aquifers, however, isn't this equivalent to the aquifer thickness? If so, does this particular correlation analysis make sense? And how might this inconsistency (meaning and hydrogeologic role of the permeable-layer thickness variable) affect the results and interpretation?

(3) Why was the topographic slope considered as an explanatory variable? Did the average slope vary significantly across the $n = 91$ aquifer systems? Note that the topography of hydrogeologic units does not necessarily track with land surface gradients. For example, the Paleozoic carbonate aquifers near the Black Hills uplift dip steeply to the east (away from outcrop areas in the Hills).

(5) line 229: I suggest a more careful wording for this subsection header – e.g., Pumping-induced drawdown “may impact” groundwater quality. The authors have not demonstrated any impacts to groundwater quality in this study, and the example provided in the text is from southeast Asia (whereas this study considered aquifer systems in the US).

REVIEWER COMMENTS

Reviewer #1 (Remarks to the Author):

Many parts of the US rely on groundwater as a source of clean drinking water and as a source of irrigation water. Aquifers that are heavily pumped and where recharge is enhanced by irrigation or managed aquifer recharge can increase the vertical movement of groundwater than would exist if these human activities were absent. However, it is a difficult task to show a relationship between pumping and the penetration depth of modern water.

The authors compiled an extensive digital dataset of aquifer boundaries and computed the fractions of modern water for wells within each aquifer. They show the depth where modern water is scarce based on the depths where 60%, 70%, and 80% of wells have small fractions of modern water. They use groundwater withdrawal data to show that pumping has marginal relationship with the depth of scarce modern.

We thank Reviewer #1 for their helpful comments. We feel that the revisions and substantial new analyses motivated by their comments improved our manuscript considerably.

Intuitively, it reasonable to expect that the vertical movement of groundwater is enhanced by pumping, however, the data and analysis is not concretely supportive of this hypothesis. I think part of this stems from the fact that some aquifers are unconfined while others are largely confined, with smaller areas of aquifer that are unconfined (Cambrian-Ordovician, Mississippi-Embayment for example). These distinctions are important because unconfined aquifers will draw water downward due to gravity drainage whereas confined aquifers are more likely to dewater clays locally. Consequently, a heavily pumped unconfined aquifer may increase the vertical penetration of modern water whereas a heavily pumped confined aquifer could actually increase the age of old groundwater in a well (see Zinn and Konikow,2007; doi:10.1029/2006WR004865). As a result, the effect of pumping on penetration depth of modern water might be easier to see in unconfined aquifers rather than by lumping them all together or differentiating them by lithology.

We found this to be an especially constructive, helpful comment. Thank you.

Before we could test the reviewer's hypothesis (that the effect of pumping on penetration depth of modern water might be easier to see in unconfined aquifers rather than by lumping them all together or differentiating them by lithology) we required an estimate of the depth to confined conditions for each of our study areas.

Therefore, we estimated the depth to confined conditions in each of our study aquifers; we added 74 new subsections within our Supplementary Information (Supplementary Notes 3.1 to

3.74). Each subsection provides up to three different sources of information used to estimate the depths at which aquifer systems transition to confined conditions:

- **(Data Source #1 – Digitization of hydrogeologic cross sections from local-scale studies)**
Two new co-authors of revised manuscript—Merhawi GebreEgziabher GebreMichael and Jobel Y. Villafañe Pagán—contributed by reviewing local-scale hydrogeologic reports and digitizing dozens of hydrogeologic cross sections. These hydrogeologic cross sections delineate confining units for many (but not all – see “Data Source #2” below) aquifer systems. Specifically, for each of our study aquifers, we digitized a hydrogeologic cross section found within a local- or regional-scale hydrogeologic report. Next, we calculated the depth to the top of the uppermost confining unit along each of the cross sections. These depths to the top of the uppermost confining unit helped us to incorporate hydrogeologic interpretations by regional hydrogeologic experts into our evaluation of the depth to the top of the uppermost confining unit. Below is one example of a hydrogeologic cross section used to estimate the depth to the uppermost confining unit (left). To the right is a map showing each of the hydrogeologic cross sections that we consulted as we completed our local-scale studies of each of our 74 study areas.

- **(Data Source #2 – Wells that the USGS has defined as tapping an unconfined or a confined aquifer)** There are several hundred thousand wells that the USGS has defined as tapping either an unconfined or confined aquifer. We downloaded this well information April 22, 2022 from <https://waterdata.usgs.gov/nwis/inventory>. Next, we evaluated the proportion of wells classified as confined versus unconfined at deeper versus shallower depths. Specifically, we grouped wells on the basis of their well depths (10 m intervals for wells with depths of shallower than 100 m; 20 m intervals for wells with depths of deeper than 100 m). Next, we calculated the fraction of wells—for which the USGS has defined the well to tap unconfined or confined conditions—that fall within a given depth interval (e.g., well depths between 10 m and 20 m) that are classified as confined. We then created plots of well depth versus the fraction of wells defined by the USGS to tap confined conditions. Last, to estimate the depth to confined conditions on the basis of these USGS wells, we determined the shallowest depth at which two criteria were met:

(i) the fraction of wells that tap confined conditions exceeds 80 % (i.e., more than 80% of wells within the depth interval are classified by the USGS as confined), *and* (ii) more than 80% of wells with depths of deeper than the depth interval are also classified as tapping confined conditions by the USGS. Here (figure below) is an example of one such plot where, in this case, the depth to confined conditions (i.e., the depth range that meets the criteria (i) and (ii) described in this paragraph) is 10-20 meters below land surface (for individual plots for each of our study aquifers see Supplementary Notes 3.1-3.74).

- (Data Source #3 – Quotes from local-scale hydrogeologic reports pertaining to the prevalence of confined conditions) We reviewed local-scale hydrogeologic studies for each of our study aquifers; we identified statements within these local-scale studies pertaining to the prevalence of confined conditions in the aquifer, and drew insights from these quotes as we created our estimate of the depth to confined conditions. For example, for the Honey Lake Valley study area, Mayo et al. (2010) state (quote): "Three groundwater systems have been identified in Honey Lake basin: (1) shallow unconfined and semiconfined (<200 m below ground surface (bgs)), (2) deep confined (>200 m bgs), and (3) geothermal" (quoting Mayo, A. L., Henderson, R. M., Tingey, D., Webber, W. (2010). Chemical evolution of shallow playa groundwater in response to post-pluvial isostatic rebound, Honey Lake Basin, California–Nevada, USA. Hydrogeology Journal, 18, 725-747). This statement was considered as we estimated the depth to confined conditions for Honey Lake Valley.

Our estimated depth to confined conditions for each of our N=74 study aquifers is included as (a) a new supplementary figure (see plot with aquifer system titles along the bottom reproduced

below), (b) a map within the supplementary information (see large map below), and (c) in the main text of our manuscript via a new figure panel.

We revised much of our original main text subsection “Deeper modern water where groundwater withdrawal rates are high” to include these new analyses of depth to confined conditions (tracked changes manuscript provided at the end of this response letter). Specifically, we added the following text to the subsection of our manuscript entitled “Deeper modern water where groundwater withdrawal rates are high”:

“It is plausible that shallow low-permeability geologic formations could limit the depth below which modern groundwater is scarce and influence the effects of pumping on vertical variability in groundwater age²⁶. We estimated the depth to confined conditions for each of our n=74 study areas by analysing (i) vertical variations in the proportion of wells defined as tapping confined aquifers by the US Geological Survey, (ii) hydrogeologic cross sections derived from local-scale studies, and (iii) statements pertaining to confined conditions from local-scale studies (see Fig. 3b; for detailed approach for estimating depth to confined conditions for each of our n=74 study aquifers see Supplementary Notes 3.1 to 3.74). We show that the depth below which modern groundwater is scarce tends to be deeper in aquifer systems characterized by thicker unconfined zones (i.e., aquifers with a relatively high (i.e., deep) depth to confined conditions; see Supplementary Note 4).

Therefore, this significant positive correlation between annual groundwater withdrawals and the depth below which modern groundwater is scarce (Fig. 3a) could arise spuriously if groundwater pumping tends to be higher in aquifer systems characterized by thick unconfined zones. To account for potential interrelationships between groundwater withdrawals and the depth to confined conditions, we completed multiple regression of the rank transforms of each explanatory variable²⁷ (Supplementary Notes 4 and 5). The resulting partial regression coefficients (β) that describe the statistical relationship between groundwater withdrawals and the depth below which modern groundwater is scarce remain positive and significant (β values range 0.29 to 0.34; all significant at Spearman P-value < 0.05; see Supplementary Table 79). This analysis suggests that our finding that modern groundwater tends to reach deeper depths in aquifer systems that are heavily pumped holds even after accounting for differences in the depth to confined conditions among our study aquifers.”

Another change we implemented in response to this reviewer comment was incorporating the recommended paper by Zinn and Konikow as a citation within our main text. Thank you for this helpful recommendation. Here is the statement in our revised manuscript that cites Zinn and Konikow (2007, which is ref. 26 in our revised manuscript):

“It is plausible that shallow low-permeability geologic formations could limit the depth below which modern groundwater is scarce and influence the effects of pumping on vertical variability in groundwater age²⁶.”

For the statistical relationship to be valid, there must also be some statistical relationship within some groups of the ensemble. The graph is not very convincing. Maybe the relationship is just a result of grouping many aquifers with different lithologies/confinement/thicknesses?

We thank Reviewer 1 for their encouragement to explore the importance of confining units. The resulting analyses of depth to confined conditions demonstrate, as Reviewer 1 successfully predicted, that depth to confinement is a key explanatory variable for the depth below which modern groundwater is scarce. We created a new Fig. 2 (main text) in an attempt to ensure that

the importance of depth to confinement is made clear in our manuscript. We also added text to our manuscript—complemented by an extensive statistical analysis in a new Supplementary Note—highlighting the importance of depth to confinement:

“We show that the depth below which modern groundwater is scarce tends to be deeper in aquifer systems characterized by thicker unconfined zones (i.e., aquifers with a relatively high (i.e., deep) depth to confined conditions; see Supplementary Note 4).”

We made another addition to our work to bring in the Reviewer’s comment; we added a new Supplementary Note that tests for ‘a statistical relationship within some groups of the ensemble’ (as recommended by the Reviewer). Specifically, we re-ran our correlations after excluding all aquifer systems where the depth to confined conditions was found to be shallower than 50 m, or shallower than 100 m (see Supplementary Note 5 entitled “*Rank correlations between groundwater withdrawals and the depth below which modern groundwater is scarce if we exclude aquifer systems with shallow depths to confined conditions*”). The correlation between groundwater withdrawals and the depth below modern groundwater is scarce remains significant even after excluding aquifer systems that have shallow confining units.

As it stands now, the statistical analysis in lines 144-163 is not that supportive of the main result and is a distraction. Also, it would be expected that the alluvial basins would be the most supportive of the author’s conclusion, but the graph alone is not enough to support that inference and the author’s do not include the statistical relationship for this set of aquifers in the text and not the table. Unfortunately, the statistical relationship between pumping and depth is not strong either. But I think it might useful to discuss the variations and reasons for deeper penetration of modern water within this aquifer type, rather than the statistical analysis in lines 144-163. It would also motivate a discussion of why the relationship is lacking in other aquifer types, perhaps due to confinement?

We thank Reviewer 1 for their comment; we have made numerous updates to our manuscript in an effort to include their recommendations in our revised manuscript.

First, we agree with Reviewer 1 that our original manuscript’s lines 144-163 were a distraction; we deleted this text.

Second, we have added a new Supplementary Note and statistical analyses to test the statistical relationship between groundwater withdrawals and the depth below which modern groundwater is scarce while taking into account the interrelationship between depth to confinement and groundwater pumping (multiple regression on rank transforms). This Supplementary Note 4 highlights that our main finding—that a moderate and significant statistical relationship exists between annual groundwater withdrawals and the depth below

which modern groundwater becomes scarce—remains robust even after accounting for differences in the depth to confined conditions among our study aquifers.

The other aspect that is difficult to assess is the effect of the assigned groundwater withdrawals on the correlation with penetration depth. The step to compute the withdrawal weight isn't clear and the area-weighting could have a large influence.

Thank you for this helpful comment. We agree that our original manuscript could have been clearer about the area-weighting method. We also agree that the area-weighting method could have a large influence, and has some major limitations.

We substantially revised our approach for estimating annual groundwater withdrawals. Specifically, we wrote a new Supplementary Note (Supplementary Note 2) detailing how we have downscaled the county-scale USGS groundwater withdrawal data. We downscaled the county-level data using two datasets: (i) we downscaled the USGS-reported groundwater irrigation withdrawal data using MODIS satellite based irrigation data at 1 km² resolution, and (ii) we downscaled, for each county individually, non-irrigation groundwater withdrawals reported by the USGS using population density data. The resulting ~1 km² datasets of groundwater withdrawals (one dataset for irrigation groundwater withdrawals, another for non-irrigation groundwater withdrawals) were used to estimate total annual groundwater withdrawals within the boundaries of each of our aquifer systems (i.e., by taking the sum of total irrigation and non-irrigation groundwater withdrawals for all grids falling within the boundaries of the aquifer system). We then divided the total annual groundwater withdrawals for each aquifer system (i.e., withdrawals expressed in m³/year) by the area of the aquifer system (in m²) to report an area-normalized withdrawal (units derived from dividing a flux (length³/time) by an area (length²) are: length per unit time). We have written an extensive Supplementary Note 2 documenting these steps, including a series of maps to help readers follow our method.

We updated the x-axis label in our main text figure (i.e., our plot of groundwater withdrawals and depth below which modern groundwater is scarce) to be clearer than we were in our original submission that the groundwater withdrawals we analyze are normalized to the area of the aquifer system. The updated x-axis is shown on the figure on the following page (i.e., Fig. 3 from the main text).

The other explanatory variables were not related to depth of scarce modern water, so either explain why they were not related or omit.

Thank you for your recommendation. We agree that these other explanatory variables were not found to be significantly correlated with the depth below which modern groundwater becomes scarce. We followed Reviewer 1's recommendation and removed discussion of these other potential explanatory variables from the main text of our manuscript so as not to distract from our main analyses and hypothesis testing.

LINE COMMENTS:

Lines 45-47: I think you mean that groundwater pumping can draw groundwater to deeper depths more rapidly than it would naturally. As it is written now, it implies the modern water wouldn't ever reach that depth but that is not true, it just would take longer in the absence of pumping.

Thank you for pointing this out. We agree. We revised our original manuscript (which included the following text: "Several local- and regional-scale studies have demonstrated that groundwater pumping can draw modern groundwater down to deeper depths than it would otherwise flow to, potentially carrying pollutants into deep wells used by municipalities and rural communities") to instead read as follows in our revised manuscript:

"Several local- and regional-scale studies have demonstrated that groundwater pumping can speed up downward flow rates and enable groundwater to reach deep depths while it is still young enough to be considered 'modern'; such pumping-induced drawdown has the potential to also draw shallow pollutants into deep wells used by municipalities and rural communities"

Lines 60-62: How do you differentiate aquifers when wells are located in areas where two aquifers are present. For example, sites in Iowa could be screened in the Glacial aquifer system or the Cambrian-Ordovician aquifer system.

Many of the study areas we explore are, as Reviewer 1 correctly points out, characterized by multiple aquifers (some shallower and some deeper). We thank Reviewer 1 for their comment and for their earlier comment recommending we characterize the depth to confinement; this proved to be highly helpful and motivated us to quantify differences in hydrostratigraphy among our study aquifers via examination of hydrogeologic cross sections for our n=74 study areas (figures in Supplementary Notes 3.1-3.74).

Lines 89-90: US drinking water wells could mean public supply wells or domestic wells. It looks like you're referring to only domestic well uses (~44 million people) and not public supply well (~110 million people). These two types of wells have very different well construction characteristics and pumping characteristics. Public supply wells are typically screened deeper in the system than domestic wells, yet this paragraph seems to imply that most people are drinking shallow, modern water and that is not true.

Thank you; this is an important point, and Reviewer 1 is correct that the vast majority of drinking water wells in this dataset are for domestic water supplies rather than public water supplies, even though the former supplies a far smaller population (44 million) than the latter (110 million). We have added the following text to be clear that most of the drinking water wells in our dataset are domestic supply wells:

"however, we stress that most drinking water wells are domestic water supply wells rather than public water supply wells, and that the latter tend to be deeper than the former."

Lines 96-99: Pumping from domestic wells isn't really the problem because they pump infrequently and at lower volumes than public wells. Pumping from public supply wells, however, can perturb the systems substantially, particularly when they're concentrated in urban areas, like the SJV for example.

We agree; thank you for pointing this out. We have revised our original manuscript to now instead read as follows (green text represent text that has been added in an effort to embed Reviewer 1's comment into our manuscript):

"Drilling domestic water wells to deeper depths may reduce the likelihood of well water contamination events in some areas^{22,23,24,25}. However, drilling deeper wells to avoid shallow contaminated aquifers may be a stopgap²¹, if pumping from nearby municipal or irrigation wells draws modern groundwater downward and jeopardizes deep groundwater quality."

Lines 144-149: Be consistent when talking about variables in a statistical context. I assume you mean that groundwater pumping = groundwater withdrawals, which was defined on line 131 and used in line 135.

Thank you. We agree that we should have been more consistent with our terminology in our original manuscript; in our original submission we incorrectly interchanged 'withdrawals' and 'pumping'. We've updated our manuscript at this location in the text (i.e., lines 144-149 of our

original manuscript highlighted by Reviewer 1 in this comment) so that we consistently use the term 'withdrawals' (to be consistent with our definition on line 131 of our original manuscript).

Lines 144-149: I do not understand this. Groundwater pumping and depth below which modern groundwater is scarce was positive and significant in previous paragraph, so why prove it again? The first sentence seems to suggest you are going to understand how groundwater pumping and the other explanatory variable are related, which may help explain why the variables aren't correlated with depth. But then the other variables are not discussed.

Thank you for your comment. We have removed mention of these other explanatory variables in order to follow the earlier Reviewer 1 comment (i.e., the earlier comment from the Reviewer recommending (quote) "*The other explanatory variables were not related to depth of scarce modern water, so either explain why they were not related or omit.*").

Lines 151-163: This does not add anything to the discussion. It might be best to exclude those aquifers from the beginning.

Thank you for this recommendation. We agree. We made the following revisions in an effort to incorporate Reviewer 1's recommendation:

(A) We added the following statement to the introduction section of our manuscript, to make clear that these n=17 aquifer systems have been excluded from further analyses:

"We excluded n=17 aquifer systems where a visual inspection of modern groundwater variations with depth suggests that our approach did not adequately capture the depth below which modern groundwater becomes scarce (Supplementary Note 1; Supplementary Figs. 1-91; Supplementary Table 1)."

(B) We have rearranged our supplementary information so that our depth vs. modern water plots are the first supplementary note (i.e., Supplementary Note 1 in the revised manuscript). We include a clear statement that the aquifers with depth vs. modern water plots that did not pass visual inspection have been excluded from further analyses; this statement reads as follows:

"These aquifer systems (where our method was deemed imperfect following visual inspection) are listed in Supplementary Table 1. We excluded these aquifer systems from further analyses and from the plots and correlations presented within the main text."

(C) We added a paragraph to our methods providing details about the excluded aquifer systems. This paragraph reads as follows:

"Third, to further scrutinize our results, we visually inspected individual plots of modern groundwater prevalence with depth for each of our study aquifers (Supplementary Figs. 1-91). We identified n=17 aquifer systems where our approach (i.e., calculating the depth below which 60%, 70% or 80% of samples contain less than 25% modern groundwater) provided an imperfect estimate of the depth below which modern groundwater becomes scarce due (a) a lack of groundwater tritium data at depth, or (b) a lack of a consistent decline in modern groundwater fractions with depth (Supplementary Table 1 and

Supplementary Figs. 4, 6, 9, 12, 16, 19, 33, 37, 44, 52, 54, 59, 71, 79, 82, 87, 88). We excluded these n=17 aquifer systems from our analyses.”

Lines 165-168: Two out of the three correlations have p-values greater than 0.05 and the rho range from 0.34 to 0.38, so these are not strong correlations.

Thank you for this comment. This text (i.e., Lines 165-168 in our original manuscript) has been deleted, as our analyses now focus on depth to confined conditions.

Lines 183-194: Please define the units of groundwater withdrawals. I don't think is millimeters per year unless you took the total and divided it by the aquifer boundaries, in which case please state that in the caption or maybe label it millimeters per square meter of aquifer per year.

We agree that we could have been clearer about the units. The Reviewer is correct that we took the total and divided it by the aquifer boundaries; we follow the Reviewer's recommendation and state that in the caption. The revised figure caption includes the following new sentence:

“Annual groundwater withdrawals were normalized by aquifer area, such that the units (mm/year) can be interpreted as all withdrawn groundwater during the year 2015 expressed as saturated layer if it were spread evenly across the study area (for study area boundaries see panel b).”

We display mm/year units as these are consistent with regional-scale hydrogeologic studies (e.g., Figure 3 of Majumdar, S., Smith, R., Butler Jr, J. J., & Lakshmi, V. Groundwater withdrawal prediction using integrated multitemporal remote sensing data sets and machine learning. *Water Resources Research*, 56, e2020WR028059 (2020). Accessed May 5, 2021 via: <https://agupubs.onlinelibrary.wiley.com/doi/full/10.1029/2020WR028059>).

We made another change to our manuscript in response to the Reviewer's comment. We updated the x-axis label in our main text figure (i.e., revised manuscript Fig. 3) to clarify that the annual groundwater withdrawals have been normalized to (i.e., divided by) aquifer system area.

Lines 207-209: This should be the last explanation not the first, particularly because it is speculative, although I agree it can be a factor. The part that is missing from this analysis, is the recharge component. Pumping alone will draw modern water downward but it becomes diminished as the head gradient decreases over time. But in places where irrigation is applied, like the central valley or high plains, or where recharge managed, like the LA Basin, the gradients can be sustained and provide an additional driving force for the deeper penetration of modern groundwater. This needs to be spelled out in the paper and it should be the second explanation for deep penetration of modern water, not the third. The first should be pumping. I think it short-circuit effects are important and are common but it needs to more study to quantify it's impact more broadly.

Thank you for these helpful recommendations to rearrange and improve this paragraph. We made two changes in response to this comment.

First, we rearranged the order of the processes we describe. Specifically, we follow the Reviewer's recommendation to list the process that we previously listed first (namely, the potential for poorly sealed wells to create conduits) to instead be listed last (following the Reviewer's comment "*This should be the last explanation not the first, particularly because it is speculative*").

Second, we agree with the Reviewer that our paper should do a better job of spelling out the importance of recharge in maintaining head gradients. In addition to our existing statements pertaining to managed aquifer recharge and artificial recharge processes associated with land uses (e.g., irrigation return flows), we added the following new sentence:

"Recharge induced by such processes can sustain high-magnitude and downward-oriented vertical hydraulic gradients and may drive modern groundwaters to deeper depths."

Lines 219-222: The fourth seems speculative but plausible.

We added the following short statement to the main text in an effort to bring the Reviewer's point into the main text:

"; this mechanism is speculative but plausible."

Lines 303-306: I thought there were two $3H_{\text{post-1953}}$ cut off values used? Which one was used as the basis for computing $F_{\text{post-1953}}$? Does the maximum $F_{\text{post-1953}}$ correspond to the minimum $3H_{\text{post-1953}}$ threshold?

Thank you for recommending we provide clarifications to our approach here. We agree that our original manuscript could have been clearer. We have added the following sentence to our methods section at the lines specified by the Reviewer (i.e. at lines 303-306 of our original manuscript):

"The maximum $F_{\text{post-1953}}$ values were determined for each sample using the minimum $^3H_{\text{post-1953}}$ values in equation 1 (i.e., entering the minimum $^3H_{\text{post-1953}}$ value into equation 1 yields the maximum $F_{\text{post-1953}}$ value for a given groundwater sample)."

Line 333: You mean groundwater withdrawals or is there another groundwater pumping dataset?

Thank you for catching this; we have corrected our manuscript to state 'withdrawals' rather than 'pumping'.

MINOR COMMENTS

Supplementary notes are not referenced in order.

Thank you. We have substantially revised our supplementary information and now assure that we present our supplementary notes in the same order that these Notes are referenced in the main text.

SUPPLEMENTAL INFORMATION COMMENTS

Give the development of a new aquifer system database, it would seem appropriate to have this published on its own and receive peer **review prior to it's use.**

We thank Reviewer 1 for their comment. Since we submitted our original manuscript the US aquifer system database has undergone peer review and has been published (GebreEgziabher, M., Jasechko, S. & Perrone, D. Widespread and increased drilling of wells into fossil aquifers in the USA. *Nat Commun* **13**, 2129 (2022). <https://doi.org/10.1038/s41467-022-29678-7>). We now reference this recent study (and we have since removed our original manuscript's supplementary note devoted to describing the aquifer system delineation process that appeared in our original supplementary information).

There are many places where one aquifer overlies another. For example, the Surficial aquifer overlies the Floridan in parts of Florida.

We agree, wholeheartedly, that there are many places where one aquifer overlies another. Our 2D study areas are treated as hydrogeologic study areas that we use to subset our database to produce depth profiles of modern groundwater. Motivated by this reviewer's recommendation to improve the way we consider vertical variations within our study areas, we completed an extensive project involving developing hydrogeologic cross sections with a standardized y-axis in order to (i) understand the depth below the land surface to the uppermost confining unit, and (ii) to support our review of the regional-scale hydrogeology for each of our study aquifer systems so that our tritium-based results can be placed into a locally relevant context (presented in an extensive new Supplementary Note 3 within the revised Supplementary Information).

How do you assign a well to the aquifer? Since this dataset is composed largely of USGS data, did you use the aquifer codes associated with each well to help differentiate aquifer assignments to wells?

We found this particular reviewer comment to be exceedingly helpful. We examined USGS definitions of wells that tap either unconfined ("Unconfined single aquifer" or "Unconfined multiple aquifer") or confined ("Confined multiple aquifers" or "Confined single aquifer") to

develop a quantitative estimate of the depth to confined conditions (figures within Supplementary Notes 3.1 to 3.74) in each of our study areas.

The other update we made to our manuscript in response to this comment was to clarify in the main text that our study areas are underlain by multiple geologic formations; we added the following statement to our revised manuscript:

“These two-dimensional aquifer system areas are underlain by multiple geologic formations of varying hydrogeologic characteristics (for hydrogeologic cross sections for each study area, see Supplementary Figs. 99-240).”

We were also motivated by this comment to examine the prevalence of modern groundwater in samples collected from wells defined as tapping a confined aquifer. We wrote a new Supplementary Note 6 that describes the spatial distribution of high-tritium groundwaters in wells tapping confined aquifers. Specifically, we found that about one-third of wells defined as tapping a confined aquifer have some (>5%) modern groundwater, highlighting that modern water reaches some wells that have been categorized as tapping one or more confined aquifers:

We wrote a new short paragraph in our main text discussion describing this new analysis; it reads as follows:

“To better understand the potential for confining units to limit the depth that modern groundwater reaches, we examined the prevalence of modern groundwater in n=1,831 groundwater samples where the well has been defined by the US Geological Survey as tapping a confined aquifer (well metadata field aqfr_type_cd specifies ‘Confined single aquifer’ or as ‘Confined multiple aquifers’). We find that about one-third of groundwater samples collected from a well that is perforated in a confined aquifer (n=592) contain more than 5% modern water, highlighting that modern water can enter wells screened in confined aquifers (Supplementary Note 7; Supplementary Fig. 244).”

We also present a new Supplementary Table 83 that provides a quantitative assessment of the proportion of wells defined as tapping a confined aquifer that have >5% modern water, ranking these aquifers from highest to lowest percentage (i.e., the highest percentage of wells defined as tapping a confined aquifer is in the Dougherty Plain and Marianna Lowlands subarea of the broader Floridan Aquifer System).

Aquifer system title	Number of well water samples collected from confined aquifer	Percent of all samples collected from a well tapping a confined aquifer that have more than 5% modern water
Dougherty Plain and Marianna Lowlands, Floridan Aquifer System	25	84% of samples from confined aquifers
Springfield Plain, Central Lowland Till Plain	29	83% of samples from confined aquifers
Ocala Uplift, Floridan Aquifer System	63	81% of samples from confined aquifers
Newcastle Till Plain, Central Lowland Till Plain	39	79% of samples from confined aquifers
Western Cambrian-Ordovician Aquifers, Northern Midwest Aquifer System	27	70% of samples from confined aquifers
Volcanic Rift Zone, Eastern Snake River Plain	12	67% of samples from confined aquifers
Balcones Fault Zone, Edwards-Trinity Aquifer System	58	64% of samples from confined aquifers
Los Angeles Basin	27	56% of samples from confined aquifers
Michigan Basin	20	55% of samples from confined aquifers
Salt Lake Valley	81	54% of samples from confined aquifers
Central Wabash and Bloomington Ridged Plain, Central Lowland Till Plain	66	50% of samples from confined aquifers
Mount Vernon Hill County, Central Lowland Till Plain	10	50% of samples from confined aquifers
Black Hills Uplift	53	43% of samples from confined aquifers
Utah Lake Valley	16	38% of samples from confined aquifers
Eastern Silurian-Devonian Aquifers, Northern Midwest Aquifer System	31	35% of samples from confined aquifers
Eastern Carrizo-Wilcox, Carrizo-Wilcox	26	35% of samples from confined aquifers
Northern High Plains, High Plains	38	32% of samples from confined aquifers
Santa Clara-Calleguas Basin	16	31% of samples from confined aquifers
Pearl and Chattahoochee Aquifer System	24	29% of samples from confined aquifers
Black Warrior River Aquifer System (Eutaw and McShan Formations and Tuscaloosa Group)	28	29% of samples from confined aquifers
Intermediate Aquifer, Floridan Aquifer System	12	25% of samples from confined aquifers
Southern High Plains, High Plains	21	24% of samples from confined aquifers
Mississippian-Silurian-Devonian Carbonates, Northern Midwest Aquifer System	56	23% of samples from confined aquifers
San Joaquin Basin, California Central Valley	15	20% of samples from confined aquifers
Central Mississippi Embayment, Mississippi Embayment	58	19% of samples from confined aquifers
Vidalia Upland, Floridan Aquifer System	17	18% of samples from confined aquifers
Peedee and Black Creek and Cape Fear Aquifers	23	17% of samples from confined aquifers
Eastern Cambrian-Ordovician Aquifers, Northern Midwest Aquifer System	12	17% of samples from confined aquifers
Bighorn Basin	14	14% of samples from confined aquifers
Eastern Mississippi Embayment, Mississippi Embayment	37	14% of samples from confined aquifers
Delmarva Peninsula, North Atlantic Coastal Plain	24	13% of samples from confined aquifers
Western Carrizo-Wilcox, Carrizo-Wilcox	26	12% of samples from confined aquifers
Tifton Upland, Floridan Aquifer System	64	11% of samples from confined aquifers
Houston-Galveston Area, Gulf Coast Regional Aquifer System	80	10% of samples from confined aquifers
Central Carrizo-Wilcox, Carrizo-Wilcox	11	9% of samples from confined aquifers
Lafayette Area, Gulf Coast Regional Aquifer System	12	8% of samples from confined aquifers
Tulare Basin, California Central Valley	14	7% of samples from confined aquifers
Upper Santa Ana Basin	18	6% of samples from confined aquifers
Tijuana-San Diego	22	5% of samples from confined aquifers
New Jersey Coastal Plain, North Atlantic Coastal Plain	29	3% of samples from confined aquifers
Lower Coastal Plain, Floridan Aquifer System	30	3% of samples from confined aquifers
Ozark Plateaus Aquifer System	30	3% of samples from confined aquifers
Western Mississippi Embayment, Mississippi Embayment	46	2% of samples from confined aquifers
Maryland Western Shores, North Atlantic Coastal Plain	23	0% of samples from confined aquifers
North Carolina and Virginia Coastal Plain, North Atlantic Coastal Plain	22	0% of samples from confined aquifers
Powder River Basin, Northern Great Plains	13	0% of samples from confined aquifers

Did you really map hundreds of aquifer systems or did you compile, digitize, georeference numerous mapped aquifers?

Thank you for your comment. We agree that coauthor M. GebreEgziabher compiled, digitized, and georeferenced numerous mapped aquifers in their recent publication that we cite as the compilation of hydrogeologic study area boundaries (GebreEgziabher, M., Jasechko, S. & Perrone, D. Widespread and increased drilling of wells into fossil aquifers in the USA. Nat Commun 13, 2129 (2022). <https://doi.org/10.1038/s41467-022-29678-7>). We deleted the following statement that appeared in our original manuscript:

“Aquifer system boundaries were delineated by georeferencing maps from hundreds of primary literature references”

Supplementary Table 7. Are these meters?

Thank you for pointing this out; yes, these values are in units of meters and we agree that our original Supplementary Table 7 was unclear about these units.

Our revised supplementary information no longer contains the original Supplementary Table 7 (as we have moved away from the classifications presented in our original manuscript to instead present depth to confined conditions, following the earlier comment from Reviewer 1). We did, however, check the Supplementary Tables within our revised Supplementary Information and ensure that units are clearly specified within these tables.

We thank Reviewer 1 for their helpful comments; we feel that our manuscript has been improved by incorporating their recommendations.

Reviewer #2 (Remarks to the Author):

This study utilized existing, publicly-available data to investigate the potential controls on modern groundwater depths within numerous aquifer systems in the USA. The authors find a significant dependence on groundwater pumping (i.e., the penetration depth of modern groundwater is greater in aquifer systems with higher overall pumping rates), as revealed by tritium concentrations in well water samples. The geospatial, mapping, and statistical analyses are impressive. The results of the study do not directly advance hydrologic understanding because the relationship between pumping and modern groundwater depth is not mechanistically explained for the various aquifer settings considered. However, this manuscript offers a valuable compilation of existing data, with novel geospatial analysis, which will be useful to the water resources research community. I recommend that the paper be published after some moderate revision. Specific comments and questions are listed below.

We thank Reviewer 2 for their thorough and helpful review. The changes we made to the manuscript in response to their recommendations improved the manuscript, in our view.

(1) As the authors are likely aware, aquifer properties (transmissivity, storage coefficients) and aquifer structure (e.g., heterogeneity, presence of confining units) are important controls on the penetration depth of modern groundwater. Although these factors were not included as explanatory variables, the grouping of aquifers by geologic setting (Figure 3) is a useful exercise that partially addresses this. However, I think the manuscript would benefit from some additional consideration of these factors, at least with added discussion and/or description of limitations. For example, note that some of the aquifer systems are multi-aquifer systems with regionally extensive confining units (e.g., Mississippi Embayment, Trinity Aquifer System, Denver Basin, and others). The presence of low-K confining units that limit vertical hydrologic connectivity would seem to be an important control on the depth of modern groundwater.

We found this comment by Reviewer 2 to be particularly motivating and beneficial. Thank you. Reviewer 1 also helpfully recommended that we do a better and more thorough job of considering not only aquifer system categories but also how hydrostratigraphy varies with depth. We agree that our original manuscript largely failed in this respect as it lacked a quantitative evaluation of the depth to confining units.

We have made substantial revisions to our revised manuscript in response to Reviewer 1's recommendation and Reviewer 2's recommendation to better characterize the spatial distribution of low-K confining units in our study areas by:

1. Rescaling and digitizing a hydrogeologic cross section from a local-scale study for all of our n=74 study areas, and interpreting the depth to the top of the uppermost confining unit from these hydrogeologic cross sections (see new Supplementary Note 3). The was a

considerable undertaking that motivated us to invite two new coauthors who helped to lead this hydrogeologic cross section digitization effort.

2. Downloading >300,000 wells where the USGS has classified the well to tap either unconfined (“Unconfined single aquifer” or “Unconfined multiple aquifer”) or confined (“Confined multiple aquifers” or “Confined single aquifer”) to estimate of the depth to confined conditions (figures within Supplementary Notes 3.1 to 3.74) by calculating the shallowest depth at which both of the following criteria are met: (a) >80% of wells within the given well depth range (e.g., grouping all wells within the study area that have depths between 80-90 m) are classified as confined by the USGS, and (b) >80% of wells with depths exceeding that of a given well depth range (e.g., all wells within the study area that have depths of >90 m)
3. Reviewing local-scale hydrogeologic studies for each of our n=74 study areas and identified statements within these local-scale reports that help to classify the depth to confined conditions. For example, for the Honey Lake Valley study area we read the following statement by Mayo et al. (2010): "Three groundwater systems have been identified in Honey Lake basin: (1) shallow unconfined and semiconfined (<200 m below ground surface (bgs)), (2) deep confined (>200 m bgs), and (3) geothermal" (quote from Mayo, A. L., Henderson, R. M., Tingey, D., Webber, W. (2010). Chemical evolution of shallow playa groundwater in response to post-pluvial isostatic rebound, Honey Lake Basin, California–Nevada, USA. *Hydrogeology Journal*, 18, 725-747). This statement was considered as we estimated the depth to confined conditions for Honey Lake Valley.

These depths to confined conditions are discussed on an aquifer by aquifer basis (i.e., for each of our n=74 study areas we wrote a Supplementary Note for that aquifer system; see Supplementary Notes 3.1-3.74).

Further, these depths to confined conditions were found to be a useful explanatory variable for the depth below which modern groundwater becomes scarce; we detail these statistical relationships in our new Supplementary Note 4. Specifically, we found that even after accounting for inter-relationships between groundwater withdrawals and the depth to confined conditions, our main finding—that the depth below which modern groundwater is scarce is correlated with annual groundwater withdrawals—remains statistically significant (see partial regression coefficients in tables in Supplementary Note 4).

(2) line 131: It seems that explanatory variable (iii) would have different meaning for the different types of aquifer systems considered in this study. For bedrock aquifers, this is more clearly interpretable as overlying regolith and sediment. For alluvial aquifers, however, isn't this equivalent to the aquifer thickness? If so, does this particular correlation analysis make sense? And how might this inconsistency (meaning and hydrogeologic role of the permeable-layer thickness variable) affect the results and interpretation?

Thank you for your helpful comment; we agree that our earlier potential explanatory variable (iii) (which was median thickness of permeable layers above bedrock in our original manuscript) may not have made sense.

Our revised manuscript now considers the depth to confined conditions instead; the depth to confined conditions was developed for each aquifer system individually after careful review of local-scale hydrogeologic reports and quantitative analysis of the proportion of wells of a given depth that have been classified as confined by the USGS (see new text within our revised Supplementary Information in, specifically, Supplementary Notes 3.1 to 3.74 (i.e., one Supplementary Note for each of our n=74 study areas)).

(3) Why was the topographic slope considered as an explanatory variable? Did the average slope vary significantly across the n = 91 aquifer systems? Note that the topography of hydrogeologic units does not necessarily track with land surface gradients. For example, the Paleozoic carbonate aquifers near the Black Hills uplift dip steeply to the east (away from outcrop areas in the Hills).

We agree that topographic slopes of hydrogeologic units do not necessarily track topographic slopes of land surfaces; we agree that the carbonate aquifers near the Black Hills are a good example of such a setting. We no longer consider topographic slope as a potential explanatory variable in our revised manuscript. Instead, we completed individual assessments of the depth to confined conditions for each of our study aquifer systems (new Supplementary Notes 3.1-3.74). The new text added to our manuscript reads as follows:

“It is plausible that shallow low-permeability geologic formations could limit the depth below which modern groundwater is scarce and influence the effects of pumping on vertical variability in groundwater age²⁶. We estimated the depth to confined conditions for each of our n=74 study areas by analysing (i) vertical variations in the proportion of wells defined as tapping confined aquifers by the US Geological Survey, (ii) hydrogeologic cross sections derived from local-scale studies, and (iii) statements pertaining to confined conditions from local-scale studies (see Fig. 3b; for detailed approach for estimating depth to confined conditions for each of our n=74 study aquifers see Supplementary Notes 3.1 to 3.74). We show that the depth below which modern groundwater is scarce tends to be deeper in aquifer systems characterized by thicker unconfined zones (i.e., aquifers with a relatively high (i.e., deep) depth to confined conditions; see Supplementary Note 4).

Therefore, this significant positive correlation between annual groundwater withdrawals and the depth below which modern groundwater is scarce (Fig. 3a) could arise spuriously if groundwater pumping tends to be higher in aquifer systems characterized by thick unconfined zones. To account for potential interrelationships between groundwater withdrawals and the depth to confined conditions, we completed multiple regression of the rank transforms of each explanatory variable²⁷ (Supplementary Notes 4 and 5). The resulting partial regression coefficients (β) that describe the statistical relationship between groundwater withdrawals and the depth below which modern groundwater is scarce remain positive and significant (β values range 0.29 to 0.34; all significant at Spearman P-value < 0.05; see Supplementary Table 79). This analysis suggests that our finding that modern groundwater tends to reach deeper depths in aquifer systems that are heavily pumped holds even after accounting for differences in the depth to confined conditions among our study aquifers.

(5) line 229: I suggest a more careful wording for this subsection header – e.g., Pumping-induced drawdown “may impact” groundwater quality. The authors have not demonstrated any impacts to groundwater quality in this study, and the example provided in the text is from southeast Asia (whereas this study considered aquifer systems in the US).

This is a good recommendation; we agree that Reviewer 2’s revision is more appropriate than our original manuscript’s subheading. We replaced the original subsection heading (which, previously, read as: “Pumping-induced drawdown impacts groundwater quality”) with Reviewer 2’s recommended version, which reads as follows in the revised manuscript:

“Pumping-induced drawdown may impact groundwater quality”

We thank Reviewer 2 for their helpful and constructive recommendations. Reviewer 2’s comments helped us improve our manuscript.

Manuscript entitled: Modern groundwater reaches deeper depths in heavily pumped aquifer systems

Submitted to: *Nature Communications* on July 10, 2022

Authors: Melissa Thaw ^a, Merhawi GebreEgziabher ^a, Jobel Y. Villafañe-Pagán ^{a,b}, Scott Jasechko ^{a,*}

^a Bren School of Environmental Science and Management, University of California, Santa Barbara, 93106, California, 93106, USA

^b Department of Geology, University of Puerto Rico, Mayagüez, 00682, Puerto Rico

Words in abstract: 139

Words in main text (including figure captions): ~~2671~~3306

Words in methods: ~~1922~~1590

Number of figures: 3

Number of tables: 0

Number of references: 5073

*** Corresponding author:**

Scott Jasechko

University of California at Santa Barbara

2400 Bren Hall, Santa Barbara, California, 93106

Email: jasechko@ucsb.edu

Abstract

Deep groundwater is an important source of drinking water, and can be preferable to shallower groundwaters where they are polluted by surface-borne contaminants. Surface-borne contaminants are disproportionately common in ‘modern’ groundwaters that are made up of precipitation that fell since the ~1950s. Some local-scale studies have suggested that groundwater pumping can draw modern groundwater downward and potentially pollute deep aquifers, but the prevalence of such pumping-induced drawdown at continental scale is not known. Here we analyse thousands of US groundwater tritium measurements to show that modern groundwater tends to reach deeper depths in heavily pumped aquifer systems. These findings imply that groundwater pumping can draw mobile surface-borne pollutants to deeper depths than they would reach in the absence of pumping. We conclude that intensive groundwater pumping can draw shallow groundwater deeper into aquifer systems, potentially endangering deep groundwater quality.

Main

Groundwater resources supply drinking water to billions of people^{1,2}. However, groundwater supplies are vulnerable to pollution from surface-borne contaminants, which can accompany precipitation as it infiltrates the land surface and percolates down to the water table³. Surface-borne contaminants are disproportionately common in groundwater that is made up of relatively recent precipitation^{4,5,6,7,8,9,10} known as ‘modern groundwater’—defined as groundwater comprised of precipitation that fell more recently than the year 1953. Because surface-borne contaminants are disproportionately common in modern groundwater, understanding the processes that control the depth that modern groundwater reaches is important to evaluate for evaluating water quality risks in shallower versus deeper wells^{11,12}.

Several local- and regional-scale studies have demonstrated that groundwater pumping can draw modern speed up downward flow rates and enable groundwater down to deeper reach deep depths than while it would otherwise flow is still young enough to potentially carrying be considered ‘modern’; such pumping-induced drawdown has the potential to also draw shallow pollutants into deep wells used by municipalities and rural communities^{13,14,15}. However, the prevalence of this pumping-induced drawdown of modern groundwater remains poorly understood, as no continent-wide study has tested if modern groundwater reaches deeper depths in places with higher groundwater pumping withdrawal rates also tend to have deeper modern groundwater.

The objective of this study is to test for statistical relationships between spatial patterns of groundwater pumping withdrawals and spatial patterns of the depth that modern groundwater reaches. To meet our objective, we analyse thousands of US groundwater tritium (³H) measurements (Supplementary-Fig. 61a) to calculate the depth below which modern groundwater is scarce in US aquifer systems (Methods subsection: Groundwater tritium data). Specifically, we calculated the fraction of each well water sample that is comprised of modern groundwater by comparing measured well water ³H activities to historical precipitation ³H time series¹⁶ (Methods subsection: Modern groundwater calculations). Elevated well water ³H activities indicate that modern groundwater is present in a well water sample¹⁷, because most of the precipitation that fell in the US after the year 1953 was artificially enriched in ³H by atmospheric thermonuclear testing^{16,18}. After completing our ³H-based calculations of the proportion of individual well

Formatted

water samples comprised of modern groundwater, we grouped wells located within the same aquifer system. Aquifer system boundaries were delineated by georeferencing maps from hundreds of primary literature references (Supplementary Note 4 for aquifer system delineation details; see Data Availability statement for a link to US aquifer geospatial data); (US aquifer geospatial data from ref.¹⁹). These two-dimensional aquifer system areas are underlain by multiple geologic formations of varying hydrogeologic characteristics (for hydrogeologic cross sections for each study area, see Supplementary Figs. 99-240). Next, for each aquifer system, we calculated the depth below which modern groundwater is scarce by calculating the shallowest depth below which most samples (60%, 70% or 80%; Fig. 42) collected from deeper wells contain ‘minimal modern groundwater’ (‘minimal modern groundwater’ defined as well water samples containing less than 25% modern groundwater; Methods). Last, we quantified topographic slopes (data: <https://ned.usgs.gov>), climate conditions¹⁸, alluvial thicknesses²⁰, and groundwater pumping rates²¹ within the boundaries of each study aquifer system (Supplementary Note 2). We then determined correlations between each potential explanatory variable—alluvial thickness, topographic slope, climatic aridity, and groundwater pumping—and the depth below which modern groundwater is scarce, accounting for potential interrelationships among the four explanatory variables themselves (Methods subsection: Calculating the depth below which modern groundwaters are scarce). We excluded n=17 aquifer systems where a visual inspection of modern groundwater variations with depth suggests that our approach did not adequately capture the depth below which modern groundwater becomes scarce (Supplementary Note 1; Supplementary Figs. 1-91; Supplementary Table 1). Last, we estimated groundwater withdrawal rates within the boundaries of each study aquifer system by downscaling county-scale groundwater withdrawal data provided by the US Geological Survey²⁰, and tested for spatial correspondence between groundwater withdrawals and the depth below which modern groundwater becomes scarce via rank regression (Methods subsection: Geospatial analyses of potential explanatory variables; see Supplementary Note 2 for further details pertaining to our calculations of annual groundwater withdrawals).

Fig. 1. Well water tritium (^3H) measurements across the contiguous United States. a) Light yellow shades represent lower tritium activities (below 1 tritium unit), orange shades represent mid-range tritium activities (1 to 10 tritium units) and red shades represent well waters with high tritium activities (exceeding 10 tritium units; 1 tritium unit is ~ 0.118 Bq/L). Light grey polygons underlying the yellow-red points are aquifer system boundaries published by GebreEgziabher et al. (ref.¹⁹; data available via <https://www.hydroshare.org/resource/d2260651b51044d0b5cb2d293d21af08/>). Panels (b-o) display hydrostratigraphy via cross sections for $n=14$ of the $n=74$ aquifer systems that we studied. The presented cross sections are based on figures and lithologic descriptions presented for the b) Boise Valley and Homedale Area within the broader Snake River Plain in ref.⁵¹, c) Black Hills Uplift in ref.⁵², d) Williston Basin in ref.⁵³, e) Eastern Dakota Aquifer System in ref.⁵⁴, f) Michigan Basin in ref.⁵⁵, g) Long Island in ref.⁵⁶, h) Castle Hayne Aquifer System in ref.⁵⁷, i) Lower Coastal Plain subarea of the broader Floridan Aquifer System in ref.⁵⁸, j) Garber-Wellington Aquifer System in ref.⁵⁹, k) Southern High Plains in ref.⁶⁰, l) Mesilla Valley in ref.⁶¹, m) Santa Clara-Calleguas Basin in ref.⁶², n) Tulare Basin subarea of the broader California Central Valley Aquifer System in ref.⁶³, and the o) Salt Lake Valley in ref.⁶⁴. Cross sections for each of our $n=74$ study aquifer systems are presented in Supplementary Figs. 99-240 (locations of each of the $n=74$ hydrogeologic cross sections we examined are displayed in Supplementary Note 8).

Modern groundwater common at shallow depths

We identified $n=9174$ aquifer systems with sufficient well water ^3H data to quantify the depth below which most (60%, 70% or 80%; Fig. 1a2a-c) groundwater samples contain minimal modern groundwater. Among our study aquifers, the median depths below which most groundwater samples contain minimal modern groundwater range from 37-7238-75 m (the range of median values derives from our three different quantitative definitions of ‘most’: 60%, 70% or 80%; Fig. 2-9%).

The depth below which modern groundwater is scarce is relatively shallow (less than ~50 m) in portions of the Gulf Coast Regional Aquifer System (Houston-Galveston and Lafayette sub-areas) and in the Northern Great Plains Aquifer System (Williston Basin and Powder River Basin sub-areas; Fig. 2). By contrast, the depth below which modern groundwater is scarce is relatively deep (greater than ~100 m) in Long Island, the Los Angeles Basin, California's Central Valley (Sacramento, San Joaquin and Tulare sub-areas), and alluvial basins in Arizona (e.g., San Pedro Basin) and California (e.g., Coachella Valley; for map with labelled aquifer system titles see Supplementary Fig. 1, and for individual plots for each aquifer system see Supplementary Figs. 8-1-91; for map with labelled aquifer system titles see Supplementary Fig. 98).

The observation that modern groundwater tends to be most common at shallow depths has important ramifications for well water quality, as surface-borne contaminants are disproportionately common in modern groundwaters^{4,5,6,7,8,9,10}. Critically, from a water quality risk perspective, we note that most US drinking water wells are perforated at relatively shallow depths where modern groundwater is most common; 55% of domestic water wells are shallower than 50 m, and 84% are shallower than 100 m (ref. ^{22,21}); however, we stress that most drinking water wells are domestic water supply wells rather than public water supply wells, and that the latter tend to be deeper than the former. Our finding that most domestic water wells are perforated at relatively shallow depths—where our ³H data suggest modern groundwaters are most common—implies that the water pumped by many domestic wells is dominated by modern groundwater, which is disproportionately likely to contain surface-borne contaminants.

Drilling domestic water wells to deeper depths may reduce the likelihood of well water contamination events in some areas²¹ areas^{22,23,24,25,26}. However, drilling deeper wells to avoid shallow contaminated aquifers may be a stopgap²³, should stopgap²¹, if pumping draw from nearby municipal or irrigation wells draws modern groundwater downward and jeopardize jeopardizes deep groundwater quality. However Nevertheless, the prevalence of pumping-induced drawdown at continental scale is not known. Therefore, we calculated correlations between groundwater pumping withdrawals and the depth below which modern groundwater is scarce (see next section entitled: Deeper modern water where pumping groundwater withdrawal rates are high²³); see Methods subsection: Geospatial analyses of potential explanatory variables).

Fig. 12. Spatial patterns of the depth below which modern groundwater is scarce among US aquifer systems. Yellow-blue shades represent the depth below which 60% (panel a), 70% (panel b) or 80% (panel c) of samples contain minimal modern groundwater (defined here as well water samples containing less than 25% modern groundwater). **a)** The median depth below which more than 60% of samples contain minimal modern groundwater among $n=9474$ studied aquifer systems (with sufficient groundwater ^3H data) is 37.38 meters, and the lower-upper quartile range is $44-89$ / $12-92$ meters. **b)** The median depth below which more than 70% of samples contain minimal modern groundwater among $n=8568$ studied aquifer systems is $55-60$ meters, and the lower-upper quartile range is $21-40$ / $5-113$ meters. **c)** The median depth below which more than 80% of samples contain minimal modern groundwater among $n=7461$ studied aquifer systems is $72-75$ meters, and the lower-upper quartile range is $32-147$ / $49-149$ meters. For a map with labels identifying aquifer systems and their generalized hydrogeologic characteristics see Supplementary Fig. 4. For more expansive versions of each panel (i.e., a, b and c), see Supplementary Figs. 2-4.98.

Formatted: Font: Times New Roman, 10 pt

Fig. 2. The depths below which modern groundwater is scarce in US aquifers. The shallow depth (top of light blue bar) represents the well depth below which more than 60% of samples contain minimal modern groundwater ('minimal modern groundwater' defined as a sample containing less than 25% modern groundwater). Circles represent well depth below which more than 70% contain minimal modern groundwater. The bottom of the dark blue bar represents the depth below which 80% of samples contain minimal modern groundwater. We plot n=85 aquifers for which we had sufficient tritium data to determine the depth below which more than 70% of samples have minimal modern groundwater. For an expanded version of this figure see Supplementary Fig. 5. For a map with labels identifying aquifer system titles see Supplementary Fig. 4.

Deeper modern water where **pumping groundwater withdrawal** rates are high

We test for statistical relationships between several explanatory variables and the depth below which modern groundwater is scarce. We completed geospatial analyses of four potential explanatory variables: (i) average topographic slope, (ii) average annual precipitation divided by potential evapotranspiration¹⁹, (iii) median thickness of permeable layers above bedrock²⁰, and (iv) annual groundwater withdrawals²¹ (see Methods subsection entitled "Geospatial analyses of potential explanatory variables").

We find a significant (Spearman P-value < 0.00101) positive correlation between annual groundwater withdrawals and the depth below which modern groundwater is scarce (Fig. 3a). Spearman rank correlation coefficients (ρ) range from $\rho = 0.3539$ to $\rho = 0.4042$ (all statistically significant at P-value < 0.00101; the range of ρ values derives from three correlation coefficients, each based on the depth below which 60%, 70% or 80% of samples contain less than 25% modern groundwater; Supplementary Table 278). Our finding suggests that modern groundwater tends to reach deeper depths in aquifer systems that are heavily pumped. By contrast, we did not find a consistently significant relationship between topographic slope, climatic aridity or alluvial thickness versus the depth below which modern groundwater becomes scarce (Supplementary Table 2).

It is plausible that shallow low-permeability geologic formations could limit the depth below which modern groundwater is scarce and influence the effects of pumping on vertical variability in groundwater age²⁶. We estimated the depth to confined conditions for each of our n=74 study areas by analysing (i) vertical variations in the proportion of wells defined as tapping confined aquifers by the US Geological Survey, (ii) hydrogeologic cross sections derived from local-scale studies, and (iii) statements pertaining to confined conditions from local-scale studies (see Fig. 3b; for detailed approach for estimating depth to confined conditions for each of our n=74 study aquifers see Supplementary Notes 3.1 to 3.74). We show that the depth below which modern groundwater is scarce tends to be deeper in aquifer systems characterized by thicker unconfined zones (i.e., aquifers with a relatively high (i.e., deep) depth to confined conditions; see Supplementary Note 4).

Therefore, this significant positive correlation between annual groundwater withdrawals and the depth below which modern groundwater is scarce (Fig. 3a) could arise spuriously if groundwater pumping tends to be higher in aquifer systems characterized by thick unconfined zones. To account for potential interrelationships between groundwater pumping and the other three explanatory variables (i.e., topographic slope, precipitation divided by potential evapotranspiration, alluvial thickness) withdrawals and the depth to confined conditions, we completed multiple regression of the rank transforms of each explanatory variable²⁷. (Supplementary Notes 4 and 5). The resulting partial regression coefficients (β) that describe the statistical relationship between groundwater pumping withdrawals and the depth below

which modern groundwater is scarce remain positive and significant (β -values range 0.40²⁹ to 0.45³⁴; all significant at Spearman P-value < 0.001⁰⁵; see Supplementary Table 3).

To further scrutinize our results, we visually inspected individual plots of modern groundwater prevalence with depth for each of our n=91 aquifers (Supplementary Figs. 8-98). We identified n=17 aquifer systems where 79). This analysis suggests that our approach (i.e., calculating the depth below which 60%, 70% or 80% of samples contain less than 25% modern groundwater) provided an imperfect estimate of the depth below which modern groundwater becomes scarce due (a) a lack of groundwater tritium data at depth, or (b) a lack of a consistent decline in modern groundwater fractions with depth (Supplementary Table 9 and Supplementary Figs. 11, 13, 16, 19, 23, 26, 40, 44, 51, 59, 61, 66, 78, 86, 89, 94 and 95). To further test the robustness of our significant correlation between groundwater pumping and the depth below which modern groundwater becomes scarce (Fig. 3), we excluded these n=17 aquifer systems and re-ran our statistical analyses; our results—after excluding these n=17 aquifer systems—still demonstrate a significant (Spearman P-value < 0.01) positive correlation between annual groundwater withdrawals and the depth below which modern groundwater is scarce (Supplementary Table 9; for results of multiple regression on rank transforms see Supplementary Table 10).

Our geostatistical analyses of n=91 aquifer systems suggest **finding** that modern groundwater tends to reach deeper depths where groundwater pumping rates are high (Fig. 3); this statistical relationship is stronger in aquifer systems that are heavily pumped holds even after accounting for differences in the depth to confined conditions among alluvial basins than among other types of aquifer systems (see triangles in our study aquifers).

Fig. 3; see Supplementary Table 4 for correlation coefficients among alluvial basins only). Our pan-US statistical analyses are consistent with local-scale research in California's Central Valley^{28,29}, where groundwater pumping draws draw young and shallow groundwater deeper into the aquifer system; our results suggest that pumping-induced drawdown is not unique to California's Central Valley and is likely occurring in other heavily pumped US aquifer systems.

Though we find that modern groundwater tends to reach deeper depths in heavily pumped aquifer systems (Fig. 3a), we emphasize the moderate strength of the correlation and the high proportion of unexplained variance (see substantial scatter in points in Fig. 3a). We also emphasize that the groundwater withdrawal data that we analyse²¹ are highly uncertain; further, the county-scale at which these data are available limit their local relevance and complicate geospatial analyses (see Methods subsection entitled "Limitations"). There is considerable room for improvement in US groundwater withdrawal data³⁰. Should better groundwater withdrawal data become available, these data would enable a better assessment of the statistical relationship between groundwater withdrawals and the depths that modern groundwaters reach.

Formatted: Font: Arial, 10 pt, Bold

Formatted: Font: Italic

Fig. 3. The depth below which modern groundwater is scarce tends to be deeper where annual groundwater withdrawals are high. a) Relationship between estimated annual groundwater withdrawals²¹ and the depth below which modern groundwater is scarce. Each point represents one aquifer system. Annual groundwater withdrawals were normalized by aquifer area, such that the units (mm/year) can be interpreted as all withdrawn groundwater during the year 2015 expressed as saturated layer if it were spread evenly across the study area (for study area boundaries see panel b). The Spearman rank correlation coefficient (ρ) of the data presented in panel a is $\rho = 0.3642$ (Spearman P-value < 0.001). The y-axis values corresponding to each coloured point are the depth below which >70% of samples have minimal (less than 25%) modern groundwater; vertical error bars extend to the depth below which >60% of samples have minimal (less than 25%) modern groundwater (shallower depth—i.e., top of grey error bar as displayed in plot) and the depth below which >80% of samples have minimal (less than 25%) modern groundwater (deeper depth—i.e., bottom of grey error bar as displayed in plot; see legend in upper-left

corner of plot). Points are colour-coded by the generalized geologic characteristics of the aquifer system (e.g., alluvial basins are upward-pointing triangles; see legend in lower right corner of plot). We only plot the n=85 We only plot the n=74 points representing the aquifer systems for which we had sufficient data to determine the depth below which >70% of samples have minimal (less than 25%) modern groundwater. For an alternate version of this figure with a linear y-axis see Supplementary Fig. 7. b) Locations of our study aquifer systems; colours of each aquifer system correspond to the generalized geologic conditions, following the identical colour scheme to panel a (see legend in lower right corner of figure). For a labelled map of each aquifer system's location and title see Supplementary Fig. 1. Points are colour coded by the estimated depth to confined conditions (see legend in panel b). The significant correlation between the depth below which modern groundwater becomes scarce and annual groundwater withdrawals remains significant when we consider potential statistical inter-relationships between groundwater withdrawals and the depth to confined conditions (Supplementary Note 4) or when we exclude aquifer systems with shallow depths to confined conditions (Supplementary Note 5). b) Map of our estimated depth to confined conditions in each of our n=74 study aquifer systems. Each polygon on the map represents one study area. Light blue colors represent shallower depths to confined conditions; darker blue shades represent deeper depths to confined conditions. Depths to confined conditions were estimated on the basis of up to three data sources: (i) USGS-defined well conditions (i.e., wells defined as tapping unconfined versus confined conditions by the US Geological Survey), (ii) digitization and evaluation of hydrogeologic cross sections derived from local-scale reports, and (iii) quotes within local-scale reports pertaining to the prevalence of confined conditions. For details on our estimated depths to confined conditions for each of our n=74 study areas, see Supplementary Note 3. For a comparison of the depth to confined conditions for each of our study areas see Supplementary Fig. 242.

Potential explanations for deep modern groundwater

While our tritium dataset cannot identify the specific mechanisms that transmit modern groundwater to deeper depths where pumping groundwater withdrawal rates are high, there are a number of potential mechanisms that may help explain our main finding—that modern groundwater tends to reach deeper depths in aquifer systems that are heavily pumped (Fig. 3)–4; Supplementary Note 6).

First, modern groundwater may be transmitted relatively rapidly to deep depths in heavily pumped aquifer systems that also host poorly sealed wells, which can create conduits that enable rapid movement of shallow modern groundwater to deep aquifers^{31,32,33,34}. Second, pumping from deep aquifers can alter vertical hydraulic gradients and speed up downward flows of shallow groundwaters^{13,35,36,37,28,31,32,33,34}; the existence of preferential flow pathways through permeable geologic structures (e.g., discontinuities in aquitards at faults or permeable intercalations³⁴ intercalations³⁵) may further enhance pumping-induced downward transport of modern groundwater³⁹. Third, groundwater³⁶. Second, the magnitude of downward-oriented vertical hydraulic gradients—which drive modern groundwater to deep depths—may be increased by artificial groundwater recharge associated with land use and water management²⁸ (e.g., infiltration of excess irrigation waters^{40,41} waters^{37,38}, leakage from urban water infrastructure⁴² infrastructure³⁹ or canals⁴³ canals⁴⁰). Spreading basins and intentional flooding have been applied via ‘managed aquifer recharge’ projects for decades in several heavily pumped aquifer systems in California and Arizona^{44,45} Arizona^{41,42}, potentially increasing downward groundwater flow rates in these areas. Fourth, Recharge induced by such processes can sustain high-magnitude and downward-oriented vertical hydraulic gradients and may drive modern groundwaters to deeper depths. Third, modern groundwaters may be present in some deep aquifers because they were intentionally injected via aquifer storage and recovery projects, but these projects are not common in many of our study areas⁴⁶ areas⁴³; this mechanism is speculative but plausible. Fourth, modern groundwater may be transmitted relatively

rapidly to deep depths in heavily pumped aquifer systems that also host poorly sealed wells, which can create conduits that enable rapid movement of shallow modern groundwater to deep aquifers^{35,44,45,46}.

To better understand the potential for confining units to limit the depth that modern groundwater reaches, we examined the prevalence of modern groundwater in n=1,831 groundwater samples where the well has been defined by the US Geological Survey as tapping a confined aquifer (well metadata field `aqfr_type_cd` specifies 'Confined single aquifer' or as 'Confined multiple aquifers'). We find that about one-third of groundwater samples collected from a well that is perforated in a confined aquifer (n=592) contain more than 5% modern water, highlighting that modern water can enter wells screened in confined aquifers (Supplementary Note 7; Supplementary Fig. 244).

Differences among our study aquifers in pumping well depths, well integrity, land use activities and managed aquifer recharge practices likely contribute the observed variability in the depth below which modern groundwaters become groundwater becomes scarce in our study areas. Identifying locally relevant mechanisms that may rapidly transmit shallow groundwater to deep depths is important to create strategies to protect deep groundwater quality.

Formatted: Font: Not Bold

Fig. 4. Schematic of some of the processes that may influence vertical distributions of modern groundwater. Artificial groundwater recharge (left side of schematic) can increase the magnitude of downward-oriented vertical hydraulic gradients and potentially drive modern groundwater to deeper depths. Artificial recharge can derive from urban waters (e.g., Tucson Basin⁶⁵), spreading basins associated with managed aquifer recharge projects (e.g., Upper Santa Ana Basin⁶⁶), excess irrigation waters (e.g., San Joaquin Basin⁶⁷), or leaky surface water conveyance infrastructure such as canals (e.g., Utah Lake

Formatted: Font: Arial, 10 pt, Bold

Valley⁶⁸). The construction of groundwater wells and their use via pumping (right side of schematic) may help modern waters enter deeper wells. For example, deep wells that have long perforated intervals may draw both shallow (disproportionately modern) and deep (disproportionately pre-modern) groundwater (e.g., Tulare Basin⁶⁹). Further, pumping from wells can alter vertical hydraulic gradients, potentially drawing shallow modern groundwater to deeper depths (e.g., Salt Lake Valley⁷⁰). The modern groundwater may move downward to deeper depths in the aquifer system via natural pore spaces (e.g., discontinuities in aquitards; e.g., Eastern Mississippi Embayment³⁶) or via conduits created by constructed wells themselves (e.g., Northern High Plains⁷¹). For further discussion and schematics of modern groundwater distributions see refs.^{50,72,73}. For a review of some of the studies that have posited one or more of these mechanisms as potential explanations for the distribution of modern groundwater see Supplementary Note 6. For a more expansive version of this figure see Supplementary Fig. 243.

Pumping-induced drawdown ~~impacts~~ may impact groundwater quality

Although our tritium data cannot identify specific transport mechanisms for each of the dozens of aquifer systems we studied, our analyses demonstrate a moderately strong statistical relationship that suggests pumping may lead modern groundwaters to reach deeper depths than they would flow to naturally (Fig. 33a).

Where modern groundwater is contaminated, pumping-induced drawdown of these groundwaters can threaten deep groundwater quality; however, we stress that many contaminants flow at considerably slower rates than the groundwater itself due to, for example, retardation via adsorption. The drawdown of modern groundwater may also have indirect impacts on groundwater quality. For example, in the Red River Delta (Vietnam), intensive pumping of deep aquifers has likely increased aqueous arsenic concentrations both directly via the drawdown of shallow arsenic-rich groundwaters to deeper depths³⁶ depths³², and also indirectly as these downward-flowing groundwaters contribute arsenic-mobilizing solutes⁴⁷. Because deep aquifers tend to require millennia to flush, contamination of deep groundwaters can be especially challenging to remedy, and some groundwater remediation technologies are not well suited for deep groundwater.

Deep groundwater is a globally important water supply, and its value is expected to grow where shallower groundwater stores and qualities are declining and deteriorating. We demonstrate that modern groundwater reaches tends to reach deeper depths in heavily pumped aquifer systems, signalling that pumping can rearrange groundwater flowpaths and impact deep groundwater quality.

Methods

Groundwater tritium data

We downloaded US well water tritium measurements from the water quality portal (<https://www.waterqualitydata.us/portal> accessed April 20, 2021). We excluded tritium measurements that were below analytical detection limits if the reported detection limit exceeded 0.8 tritium units (where 1 tritium unit equals 0.118 Bq/L). We excluded all measurements of media that were not categorized as "Groundwater":'Groundwater' (field code: "ActivityMediaSubdivisionName":'ActivityMediaSubdivisionName'). We deleted one record with a code reading "Systematic Contamination":'Contamination'. We converted negative values to zeros. We excluded one well water measurement with an unlikely "ActivityStartDate":'ActivityStartDate' value of September 28, 1900. If more than one tritium measurement was available for a single well, we analysed

only the most recent record (i.e., the measurement with the most recent date) and excluded the other measurements, to ensure each well was only counted once in our analysis.

We excluded all records that did not report a numeric well depth value (i.e., both of the following fields were blank: `WellDepthMeasure/MeasureValue` and `WellHoleDepthMeasure/MeasureValue`) or recorded a value of zero for the well depth. If a non-zero `WellDepthMeasure/MeasureValue` was available, we prioritized analysis of this well depth value; otherwise, we included the value of `WellHoleDepthMeasure/MeasureValue` as the well depth. We emphasize that both values (i.e., `WellDepthMeasure/MeasureValue` and `WellHoleDepthMeasure/MeasureValue`) are likely to be approximations because of the inherent uncertainties associated with well construction. We also highlight that our dataset does not provide information about perforated intervals for all of our wells, requiring us to analyse total well depths rather than screen depth intervals and introducing uncertainty into our analyses (see (ii) in the `Limitations` section below).

Modern groundwater calculations

We downloaded geospatial time series of estimated precipitation tritium for the contiguous US (from <https://www.sciencebase.gov/catalog/item/5af49307e4b0da30c1b44e10>; see ref.¹⁶). For each groundwater well for which we have tritium data, we identified the nearest precipitation tritium time series grid (in most cases, the well location lies within a precipitation tritium time series grid cell). Next, we decay-corrected all values in the precipitation tritium time series to determine the range of possible `net present` tritium activities relevant to a given groundwater tritium measurement (following method by ref.⁴⁸). To account for the strong likelihood that at least some hydrodynamic dispersion takes place as groundwater flows within aquifer systems, we smoothed these decay-corrected precipitation tritium time series by a five-year running average (following methods by ref.⁴⁹). We then calculated the maximum and minimum decay-corrected post-1953 precipitation tritium values from these running averages; hereafter these values are used as high and low values for the term ${}^3H_{post-1953}$. We then straightforwardly calculated the fraction of each groundwater sample comprised of ‘modern groundwater’ derived from precipitation that fell since the year 1953 ($F_{post-1953}$) following ref.⁴⁸:

$$F_{post-1953} = \frac{{}^3H_{sample} - {}^3H_{pre-1953}}{{}^3H_{post-1953} - {}^3H_{pre-1953}} \quad (1)$$

${}^3H_{sample}$ represents the measured tritium activity in the well water. ${}^3H_{post-1953}$ and ${}^3H_{pre-1953}$ represent the estimated local precipitation tritium activities¹⁶ after correcting for radioactive decay to the time of sampling. ${}^3H_{post-1953}$ represents the decay-corrected precipitation tritium for years between 1953 and the date the sample was analysed (applying the aforementioned five-year running average). ${}^3H_{pre-1953}$ represents the decay-corrected tritium activity for precipitation that fell prior to 1953 (assumed to be zero, as most of groundwater 3H samples in our dataset were analysed after multiple tritium-half-lives have elapsed since 1953, and pre-1953 precipitation tritium activities in the US were likely less than ~10 tritium units⁵⁰).

Formatted: Font color: Text 1

Calculation of ~~Calculating~~ the depths where ~~depth below which~~ modern groundwaters are abundant/scarce

We then created a binary categorization for each sample, defining each sample as either (i) containing minimal modern groundwater, defined as samples with a maximum $F_{post-1953}$ value of less than 0.25 (i.e., less than 25% of the sample is comprised of modern water), or (ii) samples with a maximum $F_{post-1953}$ value exceeding 0.25 (for map of these categorizations see Supplementary Fig. 6). ~~The maximum $F_{post-1953}$ values were determined for each sample using the minimum $^3H_{post-1953}$ values in equation 1 (i.e., entering the minimum $^3H_{post-1953}$ value into equation 1 yields the maximum $F_{post-1953}$ value for a given groundwater sample).~~

First, we grouped all groundwater tritium samples by the aquifer system that the well lies within. ~~We excluded any~~ ~~To avoid analysing the most data-sparse aquifers, we only analysed~~ aquifer systems with fewer than (a) at least $n=10$ unique wells with tritium measurements, and (b) at least a 100 m vertical offset between the total depths of the shallowest well and the deepest well for which we have groundwater tritium ~~data-~~ measurements.

Formatted: Font color: Auto
Formatted: Font color: Auto
Formatted: Font color: Auto

Second, for each well within each study aquifer, we calculated the proportion of all samples that derive from wells that are deeper than a given depth that contain minimal modern water (i.e., categorization (i) above). We did not calculate these proportions if fewer than five wells were deeper than the depth of interest (i.e., we require a minimum of five data points to calculate a value of 'the proportion of wells with deeper depths that contain minimal (i.e., maximum $F_{post-1953}$ is less than 25%) modern water'). For each aquifer system, we determined the depth at which 60%, 70%, or 80% of samples derived from wells deeper than this depth contain less than 25% modern water (these depths presented in Figs. 1 and 2).

Formatted: Font color: Text 1

~~Third, to further scrutinize our results, we visually inspected individual plots of modern groundwater prevalence with depth for each of our study aquifers (Supplementary Figs. 1-91). We identified $n=17$ aquifer systems where our approach (i.e., calculating the depth below which 60%, 70% or 80% of samples contain less than 25% modern groundwater) provided an imperfect estimate of the depth below which modern groundwater becomes scarce due (a) a lack of groundwater tritium data at depth, or (b) a lack of a consistent decline in modern groundwater fractions with depth (Supplementary Table 1 and Supplementary Figs. 4, 6, 9, 12, 16, 19, 33, 37, 44, 52, 54, 59, 71, 79, 82, 87, 88). We excluded these $n=17$ aquifer systems from our analyses.~~

Geospatial analyses of potential explanatory variables

For each study aquifer system with sufficient groundwater tritium data for analyses, we compared our estimates of 'the depth below which modern groundwater is scarce' (i.e., values in Fig. 1) ~~to four~~ potential explanatory variables: (a) average topographic slope (from <https://ned.usgs.gov>), (b) annual precipitation divided by potential evapotranspiration (from: <https://egi.aresi.community/data/global-aridity-and-pet-database/>), (c) median thickness of permeable layers above bedrock (from https://daac.ornl.gov/SOILS/guides/Global_Soil_Regolith_Sediment.html), and ~~(d)~~ ~~to two~~ potential explanatory variables: (a) annual groundwater withdrawals (from: <https://www.sciencebase.gov/catalog/item/get/5af3311be4b0da30c1b245d8>), and (b) the depth to confined conditions.

For topographic slope and annual precipitation divided by potential evapotranspiration (i.e., potential explanatory variables (a) and (b)), we calculated the mean value of all grids located inside an aquifer

system's boundaries. For the thickness of permeable layers above bedrock (i.e., potential explanatory variable (c)), we calculated the median value of all grids inside each aquifer system's boundaries (our rationale for quantifying the median rather than the mean value is that the maximum value any grid cell can have is capped at 50 m, making it challenging to calculate an average in aquifer systems where these high values exist; see Supplementary Note 2 for further details). For groundwater withdrawals (i.e., potential explanatory variable (d)), we weighted our county-scale groundwater pumping (a) To estimate groundwater withdrawals within each of our study aquifers (i.e., potential explanatory variable (a)), we weighted our county-scale groundwater withdrawal data by the proportion of the aquifer system's area covered by an individual county area (i.e., providing greater statistical weight to counties that cover more area within the aquifer system boundaries; ~~for~~ For further details see Supplementary Note 2).

(b) To estimate the depth to confined conditions within each of our study aquifers (i.e., potential explanatory variable (b)), we analysed (i) vertical variations in the proportion of wells defined as tapping confined aquifers by the USGS (depth to confined conditions estimated as the shallowest depth where both (a) the fraction of wells that tap confined conditions exceeds 80%, and (b) more than 80% of wells at deeper depths are classified as tapping confined conditions), (ii) hydrogeologic cross sections derived from local-scale studies, and (iii) statements pertaining to confined conditions from local-scale studies. For further details see Supplementary Note 3.

We then calculated rank correlation coefficients describing variability in our calculated values for the depth below which modern groundwater is scarce and these ~~four~~two explanatory variables: ~~our~~ (see Supplementary Note 4). Our calculations include independent correlations for each potential explanatory variable (Supplementary Table 78), and also a multiple regression of the rank transforms to account for potential interrelationships among the explanatory variables themselves²⁷ (Supplementary Table 79).

Delimiting aquifer systems across the US

Our analyses required a spatial database of aquifer system boundaries. We delineated aquifer system boundaries by georeferencing hundreds of maps derived from hundreds of individual studies of aquifer systems across the US (Supplementary Note 5). Specifically, we reviewed both peer-reviewed publications and also grey literature; we identified maps and study area descriptions of physiographic settings that have been treated as discrete study areas by local and regional-scale hydrogeologic studies. We georeferenced maps from these local and regional-scale hydrogeologic studies, and delineated the boundaries of these discrete study areas (referred to in this study as 'aquifers' or 'aquifer systems'). Some of our study areas are sub-areas within even more expansive aquifer systems; for example, we subdivided the High Plains Aquifer System into five sub-areas that have been defined by regional-scale studies. These 'Broader Aquifer Systems' have been recorded in our aquifer system geodatabase (see Supplementary Table 8). For a map of all of the aquifer systems that we delineated (n=440 aquifer system boundaries); see Supplementary Fig. 6.

As geologic conditions are critical to local and regional-scale hydrogeologic processes, we also compiled information about sediment and rock types within each of our study areas. For the n=91 aquifer systems for which we have sufficient well water tritium data for analyses, we reviewed locally relevant descriptions of these study areas from peer-reviewed and grey literature sources. Specifically, we identified quotes or figures from these locally relevant studies detailing hydrogeologic conditions, and

then categorized aquifer systems into various hydrogeologic typologies (e.g., “Volcanic (basalt) rocks and alluvial deposits”) based on this information. For specific quotes and figures forming the basis of our geologic categorization of each study aquifer system, see Supplementary Note 3 (for maps of our n=91 study aquifer systems and their geologic categories see Supplementary Fig. 1 and main text Fig. 3b).

Limitations

Our analyses have a number of limitations, four of which are described here.

(i) First, the groundwater ~~pumping data²¹~~ ~~withdrawal data²⁰~~ we analysed are highly uncertain and available only at county-scale. The substantial uncertainties in groundwater withdrawal data arise in part due to the lack of widespread groundwater withdrawal metering data, which limits our confidence in these withdrawal estimates (i.e., x-axis values in Fig. 33a). Further, the county-scale at which these groundwater withdrawal data are ~~reported²¹~~ ~~reported²⁰~~ complicates our interpretation of groundwater withdrawal data, ~~as many of our studied aquifer systems intersect a portion of one or more counties. Because no finer resolution US-wide and required us to downscale~~ groundwater withdrawal data are available, our area-weighted approach—see Methods subsection entitled “Geospatial analyses of potential explanatory variables”—implicitly assumes that groundwater withdrawals across each county are constant, which is a considerable oversimplification that could be avoided if finer resolution groundwater withdrawal data were available ~~to finer spatial resolution, introducing uncertainty in the process.~~ We emphasize that the groundwater ~~pumping~~ ~~withdrawal~~ estimates reported in Fig. 33a and our correlation coefficients are uncertain, and that the correlation coefficients we present would differ were better groundwater ~~pumping~~ ~~withdrawal~~ data to become available.

(ii) Second, our evaluation of the depths below which modern groundwater is scarce is limited by the lack of perforated interval data for some of the wells where groundwater tritium measurements have been made. Our calculations are thus necessarily limited to total well depth data, which are, if anything, an overestimate of the depth of the groundwater sampled by the well. Wells that are deeper than the total depths that modern groundwater penetrates may pump modern groundwater because some of these wells will be perforated at shallower depths where modern groundwater is more abundant, potentially leading to overestimations of the depth below which modern groundwaters penetrate in some well with extensive perforated intervals.

(iii) Third, we cannot evaluate the depths at which groundwater withdrawals take place, as the ~~pumping~~ data we ~~analyze²¹~~ ~~analyse²⁰~~ are two-dimensional data products that do not contain information about the depths at which pumping rates are higher versus lower. However, all of our study aquifers have at least some recorded ~~wells²³~~ ~~wells²¹~~ with depths that exceed the depth below which modern groundwater is scarce, implying a potential for groundwater ~~pumping~~ ~~withdrawals~~ to play a role in determining the depth at which modern groundwater becomes scarce.

~~(iv) Fourth, although our analyses represent the most densely distributed groundwater tritium available, measurements are sparse in some of the studied aquifer systems (Supplementary Fig. 6), meaning we cannot confirm that the available groundwater tritium measurements capture 3D modern groundwater variations adequately in all of our study areas (see Supplementary Fig. 6 for a map of the density of tritium measurement locations within delineated aquifer systems). To avoid analysing the most data-sparse aquifers, we only analysed aquifer systems with (a) at least n=10 tritium measurements, and~~

(b) at least a 100 m vertical offset between the total depths of the shallowest well and the deepest well for which we have groundwater tritium measurements. We also completed a thorough visual inspection of all plots (Supplementary Figs. 8–98). We identified a minority of aquifer systems (n=17 of n=91) where the approach we used to estimate the depth below which modern groundwater becomes scarce provided imperfect results (Supplementary Note 6); however, the significant (Spearman P value < 0.01) statistical relationship between groundwater pumping and the depth below which modern groundwater becomes scarce remained significant even after excluding these n=17 aquifer systems (Supplementary Tables 9 and 10).

(iv) Fourth, we analyse groundwater withdrawals for the year 2015 but acknowledge that groundwater pumping that took place prior to the year 2015 has also likely contributed to the depth distribution of modern groundwater.

References

1. Shah, T. Groundwater and human development: challenges and opportunities in livelihoods and environment. *Water Science and Technology* **51**, 27–37 (2005).
2. Margat, J. & van der Gun, J. Groundwater around the world: A Geographic Synopsis: geographic synopsis. CRC Press (2013).
3. Foster, S.S.D. & Chilton, P.J. Groundwater: the processes and global significance of aquifer degradation. *Philosophical Transactions of the Royal Society of London. Series B: Biological Sciences* **358**, 1957–1972 (2003).
4. Baran, N., Mouvet, C. & Négrel, P. Hydrodynamic and geochemical constraints on pesticide concentrations in the groundwater of an agricultural catchment (Brévilles, France). *Environmental Pollution* **148**, 729–738 (2007).
5. Brown, K. B., McIntosh, J. C., Rademacher, L. K. & Lohse, K. A. Impacts of agricultural irrigation recharge on groundwater quality in a basalt aquifer system (Washington, USA): a multi-tracer approach. *Hydrogeology Journal* **19**, 1039–1051 (2011).
6. Qin, D., Qian, Y., Han, L., Wang, Z., Li, C. & Zhao, Z. Assessing impact of irrigation water on groundwater recharge and quality in arid environment using CFCs, tritium and stable isotopes, in the Zhangye Basin, Northwest China. *Journal of Hydrology* **405**, 194–208 (2011).
7. Pang, Z., Yuan, L., Huang, T., Kong, Y., Liu, J. & Li, Y. Impacts of human activities on the occurrence of groundwater nitrate in an alluvial plain: a multiple isotopic tracers approach. *Journal of Earth Science* **24**, 111–124 (2013).
8. Han, D., Cao, G., McCallum, J. & Song, X. (2015). Residence times of groundwater and nitrate transport in coastal aquifer systems: Daweijia area, northeastern China. *Science of the Total Environment* **538**, 539–554 (2015).
9. Visser, A., Moran, J. E., Hillegonds, D., Singleton, M.J., Kulongoski, J.T., Belitz, K. & Esser, B.K. Geostatistical analysis of tritium, groundwater age and other noble gas derived parameters in California. *Water Research* **91**, 314–330 (2016).
10. El Mountassir, O., Bahir, M., Ouazar, D., Chehbouni, A. & Carreira, P.M. Temporal and spatial assessment of groundwater contamination with nitrate using nitrate pollution index (NPI), groundwater pollution index (GPI), and GIS (case study: Essaouira basin, Morocco). *Environmental Science and Pollution Research* 1–18 (2021).

11. Zhang, X., Xu, Z., Sun, X., Dong, W. & Ballantine, D. Nitrate in shallow groundwater in typical agricultural and forest ecosystems in China, 2004–2010. *Journal of Environmental Sciences* **25**, 1007–1014 (2013).
12. Egbueri, J. C. & Mgbenu, C. N. Chemometric analysis for pollution source identification and human health risk assessment of water resources in Ojoto Province, southeast Nigeria. *Applied Water Science* **10**, 1–18 (2020).
13. Kagabu, M., Shimada, J., Delinom, R., Nakamura, T. & Taniguchi, M. Groundwater age rejuvenation caused by excessive urban pumping in Jakarta area, Indonesia. *Hydrological Processes* **27**, 2591–2604 (2013).
14. Han, D.M., Song, X.F., Currell, M.J., Yang, J.L. & Xiao, G.Q. Chemical and isotopic constraints on evolution of groundwater salinization in the coastal plain aquifer of Laizhou Bay, China. *Journal of Hydrology* **508**, 12–27 (2014).
15. Taufiq, A., Hosono, T., Ide, K., Kagabu, M., Iskandar, I., Effendi, A.J. & Hutasoit, L.M. Shimada, J. Impact of excessive groundwater pumping on rejuvenation processes in the Bandung basin (Indonesia) as determined by hydrogeochemistry and modeling. *Hydrogeology Journal* **26**, 1263–1279 (2018).
16. Michel, R.L., Jurgens, B.C. & Young, M.B. Tritium deposition in precipitation in the United States, 1953–2012. US Geological Survey Scientific Investigations Report 2018-5086, 19 pp. <https://pubs.er.usgs.gov/publication/sir20185086> (USGS, 2018).
17. Gleeson, T., Befus, K.M., Jasechko, S., Luijendijk, E. & Cardenas, M.B. The global volume and distribution of modern groundwater. *Nature Geoscience* **9**, 161–167 (2016).
18. Doney, S.C., Glover, D.M. & Jenkins, W.J.A model function of the global bomb tritium distribution in precipitation, 1960–1986. *Journal of Geophysical Research: Oceans* **97**, 5481–5492 (1992).
19. ~~Zomer, R. J., Trabucco, G.~~ ~~GebreEgziabher, M., Jasechko, S. & Perrone, D.~~ Widespread and increased drilling of wells into fossil aquifers in the USA. *Nature Communications* **13**, 2129 <https://doi.org/10.1038/s41467-022-29678-7> (2022).
19. ~~A., Bossio, D.A., van Straaten, G. & Verchot, L.V.~~ Climate change mitigation: a spatial analysis of global land suitability for clean development mechanism afforestation and reforestation. *Agriculture, Ecosystems & Environment* **126**, 67–80 (2008).
20. Pelletier, J.D., Broxton, P.D., Hazenberg, P., Zeng, X., Troeh, P.A., Niu, G.Y., Williams, Z., Brunke, M.A. & Gochis, D. (2016). A gridded global data set of soil, intact regolith, and sedimentary deposit thicknesses for regional and global land surface modeling. *Journal of Advances in Modeling Earth Systems* **8**, 41–65 (2016).
21. ~~20.~~ Dieter, C.A. et al. Estimated use of water in the united states in 2015. US Geological Survey Circular 1441, <https://doi.org/10.3133/cir1441> (USGS, 2018).
22. ~~21.~~ Perrone, D. & Jasechko, S. Deeper well drilling an unsustainable stopgap to groundwater depletion. *Nature Sustainability* **2**, 773–782 (2019).
23. ~~22.~~ Clark, B.R., Hart, R.M. & Gurdak, J.J. Groundwater availability of the Mississippi Embayment. US Geological Survey Professional Paper 1785, 62 pp. <https://pubs.usgs.gov/pp/1785/> (USGS, 2011).
24. ~~23.~~ Zhang, X., Xu, Z., Sun, X., Dong, W. & Ballantine, D. Nitrate in shallow groundwater in typical agricultural and forest ecosystems in China, 2004–2010. *Journal of Environmental Sciences* **25**, 1007–1014 (2013).

- ~~25.24.~~ Egbueri, J. C. & Mgbenu, C. N. Chemometric analysis for pollution source identification and human health risk assessment of water resources in Ojoto Province, southeast Nigeria. *Applied Water Science* **10**, 1–18 (2020).
- ~~26.25.~~ Wirmvem, M.J., Ohba, T., Nche, L.A., Kamtchueng, B.T., Kongnso, W.E., Mimba, M.E., Bafon, T.G., Yaguchi, M., Takem, G.E., Fantong, W.Y. & Ako, A.A. Effect of diffuse recharge and wastewater on groundwater contamination in Douala, Cameroon. *Environmental Earth Sciences* **76**, 1–23 (2017).
- ~~26.~~ Zinn, B. A., & Konikow, L.F. Potential effects of regional pumpage on groundwater age distribution. *Water Resources Research* **43**, W06418 (2007).
27. Iman, R. L. & Conover, W. J. The use of the rank transform in regression. *Technometrics* **21**, 499–509 (1979).
28. Jagucki, M.L., Jurgens, B.C., Burow, K.R. & Eberts, S.M. Assessing the vulnerability of public-supply wells to contamination—Central Valley aquifer system near Modesto, California: US Geological Survey Fact Sheet 2009–3036, 6 pp. <https://pubs.usgs.gov/fs/2009/3036/> (USGS, 2009).
29. Levy, Z.F., Jurgens, B.C., Burow, K.R., Voss, S.A., Faulkner, K.E., Arroyo-Lopez, J.A. & Fram, M.S. Critical aquifer overdraft accelerates degradation of groundwater quality in California's Central Valley during drought. *Geophysical Research Letters* **48**, e2021GL094398 (2021).
30. Perrone, D., Hornberger, G., van Vliet, O. & van der Velde, M. A review of the United States' past and projected water use. *JAWRA Journal of the American Water Resources Association* **51**, 1183–1191 (2015).
- ~~21.1.~~ Santi, P.M., McCray, J.E. & Martens, J.L. Investigating cross contamination of aquifers. *Hydrogeology Journal* **14**, 51–68 (2006).
- ~~32.1.~~ Koh, E.H., Kaown, D., Mayer, B., Kang, B.R., Moon, H.S. & Lee, K.K. Hydrogeochemistry and isotopic tracing of nitrate contamination of two aquifer systems on Jeju Island, Korea. *Journal of Environmental Quality* **41**, 1835–1845 (2012).
- ~~23.1.~~ Han, D., Currell, M.J. & Cao, G. Deep challenges for China's war on water pollution. *Environmental Pollution* **218**, 1222–1233 (2016).
- ~~34.1.~~ Shuler, C.K., Dulai, H., DeWees, R., Kirs, M., Glenn, C.R. & El-Kadi, A. I. Isotopes, microbes, and turbidity: A multi-tracer approach to understanding recharge dynamics and groundwater contamination in a Basaltic Island Aquifer. *Groundwater Monitoring & Remediation* **39**, 20–35 (2019).
- ~~35.31.~~ Lapworth, D.J., Das, P., Shaw, A., Mukherjee, A., Civil, W., Petersen, J.O., Goody, D.C., Wakefield, O., Finlayson, A., Krishan, G. & Sengupta, P. Deep urban groundwater vulnerability in India revealed through the use of emerging organic contaminants and residence time tracers. *Environmental Pollution* **240**, 938–949 (2018).
- ~~36.32.~~ Winkel, L. H., Trang, P.T.K., Lan, V.M., Stengel, C., Amini, M., Ha, N.T., Viet, P.H., Berg, M. Arsenic pollution of groundwater in Vietnam exacerbated by deep aquifer exploitation for more than a century. *Proceedings of the National Academy of Sciences* **108**, 1246–1251 (2011).
- ~~37.33.~~ Khan, M.R., Koneshloo, M., Knappett, P.S., Ahmed, K.M., Bostick, B.C., Mailloux, B.J., Mozumder, R.H., Zahid, A., Harvey, C.F., Van Geen, A. & Michael, H.A. Megacity pumping and preferential flow threaten groundwater quality. *Nature Communications* **7**, 12833 (2016).

- ~~38.~~34. Cognac, K. E. & Ronayne, M. J. Changes to inter-aquifer exchange resulting from long-term pumping: implications for bedrock groundwater recharge. *Hydrogeology Journal* **28**, 1359–1370 (2020).
35. Santi, P.M., McCray, J.E. & Martens, J.L. Investigating cross-contamination of aquifers. *Hydrogeology Journal* **14**, 51–68 (2006).
- ~~39.~~36. Kingsbury, J.A., Barlow, J.R., Jurgens, B.C., McMahon, P.B. & Carmichael, J.K. Fraction of young water as an indicator of aquifer vulnerability along two regional flow paths in the Mississippi embayment aquifer system, southeastern USA. *Hydrogeology Journal* **25**, 1661–1678 (2017).
- ~~40.~~37. Scanlon, B.R., Reedy, R.C., Stonestrom, D.A., Prudic, D.E. & Dennehy, K.F. Impact of land use and land cover change on groundwater recharge and quality in the southwestern US. *Global Change Biology* **11**, 1577–1593 (2005).
- ~~41.~~38. Ochoa, C.G., Fernald, A.G., Guldán, S.J., Tidwell, V.C. & Shukla, M.K. Shallow aquifer recharge from irrigation in a semiarid agricultural valley in New Mexico. *Journal of Hydrologic Engineering* **18**, 1219–1230 (2013).
- ~~42.~~39. Wakode, H.B., Baier, K., Jha, R. & Azzam, R. Impact of urbanization on groundwater recharge and urban water balance for the city of Hyderabad, India. *International Soil and Water Conservation Research* **6**, 51–62 (2018).
- ~~43.~~40. Meredith, E. & Blais, N. Quantifying irrigation recharge sources using groundwater modeling. *Agricultural Water Management* **214**, 9–16 (2019).
- ~~44.~~41. Dillon, P., Stuyfzand, P., Grischek, T., Lluria, M., Pyne, R.D.G., Jain, R.C., Bear, J., Schwarz, J., Wang, W., Fernandez, E. & Stefan, C. Sixty years of global progress in managed aquifer recharge. *Hydrogeology Journal* **27**, 1–30 (2019).
- ~~45.~~42. Scanlon, B.R., Reedy, R.C., Faunt, C.C., Pool, D. & Uhlman, K. Enhancing drought resilience with conjunctive use and managed aquifer recharge in California and Arizona. *Environmental Research Letters* **11**, 035013 (2016).
- ~~46.~~43. Stefan, C. & Ansems, N. Web-based global inventory of managed aquifer recharge applications. *Sustainable Water Resources Management* **4**, 153–162 (2018).
44. Koh, E.H., Kaown, D., Mayer, B., Kang, B.R., Moon, H.S. & Lee, K.K. Hydrogeochemistry and isotopic tracing of nitrate contamination of two aquifer systems on Jeju Island, Korea. *Journal of Environmental Quality* **41**, 1835–1845 (2012).
45. Han, D., Currell, M.J. & Cao, G. Deep challenges for China's war on water pollution. *Environmental Pollution* **218**, 1222–1233 (2016).
46. Shuler, C.K., Dulai, H., DeWees, R., Kirs, M., Glenn, C.R. & El-Kadi, A. I. Isotopes, microbes, and turbidity: A multi-tracer approach to understanding recharge dynamics and groundwater contamination in a Basaltic Island Aquifer. *Groundwater Monitoring & Remediation* **39**, 20–35 (2019).
47. Berg, M., Trang, P.T.K., Stengel, C., Buschmann, J., Viet, P.H., Van Dan, N., Giger, W. & Stüben, D. Hydrological and sedimentary controls leading to arsenic contamination of groundwater in the Hanoi area, Vietnam: The impact of iron-arsenic ratios, peat, river bank deposits, and excessive groundwater abstraction. *Chemical Geology* **249**, 91–112 (2008).
48. Jasechko, S. Partitioning young and old groundwater with geochemical tracers. *Chemical Geology* **427**, 35–42 (2016).
49. Jasechko, S., Perrone, D., Befus, K.M., Cardenas, M.B., Ferguson, G., Gleeson, T., Luijendijk, E., McDonnell, J.J., Taylor, R.G., Wada, Y. ~~and~~ & Kirchner, J.W. Global aquifers dominated by

Formatted: Font: Italic

- fossil groundwaters but wells vulnerable to modern contamination. *Nature Geoscience* **10**, 425–429 (2017).
50. Jasechko, S. Global isotope hydrogeology — Review. *Reviews of Geophysics* **57**, 835–965 (2019).
 51. Lindholm, G. F. Summary of the Snake River Plain regional aquifer-system analysis in Idaho and eastern Oregon. U.S Geological Survey Professional Paper 1408-A. 59 pp. Accessed March 1, 2021 via <https://pubs.usgs.gov/pp/1408a/report.pdf> (1996).
 52. Driscoll, D. G., Carter, J. M., Williamson, J. E. & Putnam, L. D. Hydrology of the Black Hills area, South Dakota. US Geological Survey Water-Resources Investigations Report 2002-4094, 158 pp. Accessed February 16, 2021 from <https://pubs.usgs.gov/wri/wri024094/pdf/wri024094.pdf> (2002).
 53. Long, A.J., Thamke, J.N., Davis, K.W. & Bartos, T.T. Groundwater availability of the Williston Basin, United States and Canada. US Geological Survey Professional Paper 1841, 42 p., Accessed February 16, 2021 from <https://doi.org/10.3133/pp1841> (2018).
 54. Prior, J.C., Boekhoff, J.L., Howes, M.R., Libra, R.D. & VanDorpe, P.E. Iowa's groundwater basics. Iowa Department of Natural Resources Report, 92 pp. Accessed November 29, 2021 from https://s-iihr34.iuhr.uiowa.edu/publications/uploads/2014-08-24_08-08-21_es-06.pdf (2003).
 55. Westjohn, D.B. & Weaver, T.L. Hydrogeologic framework of the Michigan Basin regional aquifer system. US Geological Survey Professional Paper 1418, 55 pp. Accessed November 29, 2021 from <https://pubs.usgs.gov/pp/1418/report.pdf> (1998).
 56. Smolensky, D.A., Buxton, H.T. & Shernoff, P.K. Hydrologic framework of Long Island, New York. U.S. Geological Survey Hydrologic Atlas 709, 3 plates, Accessed June 15, 2022 from <https://pubs.usgs.gov/ha/709/plate-1.pdf> (1990).
 57. Winner Jr., M.D. & Coble, R.W. Hydrogeologic framework of the North Carolina Coastal Plain aquifer system. US Geological Survey Report 87-690, 167 pp. Accessed April 1, 2021 from <https://pubs.usgs.gov/of/1987/0690/report.pdf> (1989).
 58. Aucott, W.R. Hydrology of the Southeastern Coastal Plain Aquifer System in South Carolina and Parts of Georgia and North Carolina. U.S Geological Survey Professional Paper 1410-E, 95 pp. Accessed March 31, 2021 from <https://pubs.usgs.gov/pp/1410e/report.pdf> (1996).
 59. Mashburn, S.L., Ryter, D.W., Neel, C.R., Smith, S.J. & Correll, J.S. Hydrogeology and simulation of ground-water flow in the Central Oklahoma (Garber-Wellington) Aquifer, Oklahoma, 1987 to 2009, and simulation of avail-able water in storage, 2010–2059. US Geological Survey Scientific Investigations Report 2013–5219, 92 pp. Accessed May 14, 2021 from https://pubs.usgs.gov/sir/2013/5219/pdf/sir20135219_v2.0.pdf (2014).
 60. Blandford, T. N., Kuchanur, M., Standen, A., Ruggiero, R., Calhoun, K. C., Kirby, P. & Shah, G. Groundwater Availability Model of the Edwards-Trinity (High Plains) Aquifer in Texas and New Mexico. Texas Water Development Board. Accessed June 20, 2022 from <https://www.twdb.texas.gov/groundwater/aquifer/minors/edwards-trinity-high-plains.asp> (2008).
 61. Robertson, A.J., Matherne, A.M., Pepin, J.D., Ritchie, A.B., Sweetkind, D.S., Teeple, A.P., Granados-Olivas, A., García-Vásquez, A.C., Carroll, K.C., Fuchs, E.H. & Galanter, A.E. Mesilla/Conejos-Médanos Basin: US-Mexico Transboundary Water Resources. *Water* **14**, 134 (2022).
 62. Hanson, R.T., Martin, P. & Koczot, K.M. Simulation of ground-water/surface-water flow in the Santa Clara-Calleguas ground-water basin, Ventura County, California. *Water-Resources*

- [Investigations Report 2002-4136, 172 pp. Accessed March 20, 2021 from https://pubs.usgs.gov/wri/wri024136/wri024136.pdf \(2002\).](https://pubs.usgs.gov/wri/wri024136/wri024136.pdf)
63. Page, R. W. & Balding, G. O. Geology and quality of water in the Modesto-Merced area, San Joaquin Valley, California, with a brief section on hydrology. US Geological Survey Water-Resources Investigations Report 73-6, 77 pp. Accessed June 16, 2022 via [https://pubs.er.usgs.gov/publication/wri736 \(1973\).](https://pubs.er.usgs.gov/publication/wri736)
 64. Manning, A.H. & Solomon, D.K. An integrated environmental tracer approach to characterizing groundwater circulation in a mountain block. *Water Resources Research* **41**, W12412, doi:10.1029/2005WR004178 (2005).
 65. Carlson, M. A., Lohse, K. A., McIntosh, J. C. & McLain, J. E. Impacts of urbanization on groundwater quality and recharge in a semi-arid alluvial basin. *Journal of Hydrology* **409**, 196–211 (2011).
 66. Hamlin, S.N., Belitz, K. & Johnson, T. Occurrence and Distribution of Volatile Organic Compounds and Pesticides in Ground Water in Relation to Hydrogeologic Characteristics and Land Use in the Santa Ana Basin, Southern California. US Geological Survey Scientific Investigations Report 2005-5032, 49 pp. Accessed May 31, 2022 via [https://pubs.usgs.gov/sir/2005/5032/sir2005-5032.pdf \(2005\).](https://pubs.usgs.gov/sir/2005/5032/sir2005-5032.pdf)
 67. Bennett, G.L., Fram, M.S., Belitz, K. & Jurgens, B.C. Status and Understanding of Groundwater Quality in the Northern San Joaquin Basin, 2005: California GAMA Priority Basin Project. US Geological Survey Scientific Investigations Report 2010–5175, 96 pp. Accessed May 31, 2022 via [https://pubs.usgs.gov/sir/2010/5175/pdf/sir20105175.pdf \(2010\).](https://pubs.usgs.gov/sir/2010/5175/pdf/sir20105175.pdf)
 68. Cederberg, J.R., Gardner, P.M. & Thiros, S.A. Hydrology of Northern Utah Valley, Utah County, Utah, 1975–2005. US Geological Survey Scientific Investigations Report 2008–5197, 128 pp. Accessed May 31, 2022 via [https://pubs.usgs.gov/sir/2008/5197/pdf/sir2008-5197.pdf \(2008\).](https://pubs.usgs.gov/sir/2008/5197/pdf/sir2008-5197.pdf)
 69. Hansen, J.A., Jurgens, B.C. & Fram, M.S. Quantifying anthropogenic contributions to century-scale groundwater salinity changes, San Joaquin Valley, California, USA. *Science of the Total Environment* **642**, 125–136 (2018).
 70. Thiros, S.A. & Manning, A.H. Quality and sources of ground water used for public supply in Salt Lake Valley, Salt Lake County, Utah, 2001. US Geological Survey Water-Resources Investigations Report 2003-4325, 24 pp. Accessed May 31, 2022 via [https://pubs.usgs.gov/pp/1781/pdf/pp1781_section2.pdf \(2003\).](https://pubs.usgs.gov/pp/1781/pdf/pp1781_section2.pdf)
 71. Landon, M. K., Jurgens, B.C., Katz, B.G., Eberts, S.M., Burow, K.R. & Crandall, C.A. Depth-dependent sampling to identify short-circuit pathways to public-supply wells in multiple aquifer settings in the United States. *Hydrogeology Journal* **18**, 577–593 (2010).
 72. Ferguson, G., Cuthbert, M. O., Befus, K., Gleeson, T., & McIntosh, J. C. (2020). Rethinking groundwater age. *Nature Geoscience* **13**, 592-594.
 73. Jurgens, B.C., Faulkner, K., McMahon, P.B., Hunt, A. G., Casile, G., Young, M.B., Belitz, K. (2022). Over a third of groundwater in USA public-supply aquifers is Anthropocene-age and susceptible to surface contamination. *Communications Earth & Environment*, **3**, 1-9.

Acknowledgments. This material is based upon work supported by the National Science Foundation under Grant No. EAR-2048227. This research was supported by funding from the Zegar Family Foundation. This work was supported by funding from California Institute for Water Resources (CIWR) Water Research Program. The authors acknowledge the Jack and Laura Dangermond Preserve, the Point Conception Institute, and the Nature Conservancy for their support of this research.

Author information – Affiliations. ^a Bren School of Environmental Science and Management, University of California, Santa Barbara, 93106, California, 93106, USA ^b Department of Geology, University of Puerto Rico, Mayagüez, 00682, Puerto Rico — ^a Melissa Thaw, ^a Merhawi GebreEgziabher, ^{a,b} Jobel Y. Villafañe-Pagán, ^a Scott Jasechko

▲ **Contributions.** M.T. and S.J. contributed equally to the manuscript. M.T. and S.J. co-generated hypotheses, co-developed methods, and co-wrote the manuscript. M.T. and S.J. calculated the depth below which modern groundwater is scarce. S.J. delineated aquifer system boundaries. M.G. digitized hydrogeologic cross sections and completed the literature review to categorize geologic formations (e.g., as aquifers or as aquitards). J.Y. V-P. georeferenced maps from reviewed literature to approximate the location of each hydrogeologic cross section. S.J. estimated the depth to confined conditions for each study area.

Corresponding author. Correspondence to Scott Jasechko (jasechko@ucsb.edu)

Ethics declarations. The authors declare no competing interests.

Data Availability. Aquifer system boundaries are presented as a Supplementary Data File (shapefile) and can be accessed via the following publicly available data repository: *[a link to the geospatial dataset (posted to a public data sharing repository e.g., Dataverse, FigShare) will be made available following our work being accepted for publication. We are completing a number of distinct analyses that make use of these spatial data, and it is possible that another paper that describes these spatial data may be published before this work; should this situation arise, we will update our manuscript and supplementary information to cite that paper (rather than describing the spatial data here). As these spatial data have not been published at the time of submission (February 2022), we detail the literature sources and steps used to generate the aquifer database in the Supplementary Information, and provide access to the data for peer review purposes only — via the following confidential hyperlink: <https://app.box.com/s/o7vot58eja1452651ivj8j3fgoxbxvjf> Aquifer system boundaries from ref.¹⁹ are available to download via CUAHSI HydroShare: <https://www.hydroshare.org/resource/d2260651b51044d0b5cb2d293d21af08/>. Groundwater withdrawal estimates (county-scale) are available via <https://pubs.er.usgs.gov/publication/cir1441>. Well water tritium data are available via <https://www.waterqualitydata.us/portal>.*

Code availability. The analyses presented here do not depend on specific code. Our approach can be reproduced following the procedures described in the Methods.

Formatted: Font: Italic

Supplementary Information

Title: Modern groundwater reaches deeper depths in heavily pumped aquifer systems

Thaw, M. ^{a,*}, GebreEgziabher, M. ^a, Villafañe-Pagán, J.Y. ^{a,b}, Jasechko, S. ^{a,*}

^a Bren School of Environmental Science and Management, University of California, Santa Barbara, California, USA

^b Department of Geology, University of Puerto Rico, Mayagüez, 00682, Puerto Rico

* authors contributed equally

Table of contents

Supplementary Note 1. Supplementary results: Well depth versus modern groundwater plots for study aquifers	5
Supplementary Note 2. Geospatial analyses of explanatory variables: Estimating groundwater withdrawals within each study area	332
Supplementary Note 3. Classified geologic typologies for study aquifers: Regional hydrogeology and hydrostratigraphy of study aquifer systems	344
Supplementary Note 4. Delineating aquifer boundaries by review of local studies	58
3.1 Sacramento Basin, California Central Valley	365
3.2 San Joaquin Basin, California Central Valley	368
3.3 Tulare Basin, California Central Valley	371
3.4 Eastern Carrizo-Wilcox, Carrizo-Wilcox	374
3.5 Eagle Valley, Carson River Basin	377
3.6 Central Wabash and Bloomington Ridged Plain, Central Lowland Till Plain	379
3.7 Palouse Slope, Columbia Plateau Regional Aquifer System	382
3.8 Umatilla Basin and Horse Heaven Hills, Columbia Plateau Regional Aquifer System	384
3.9 Yakima Basin, Columbia Plateau Regional Aquifer System	387
3.10 Stockton Plateau, Edwards-Trinity Aquifer System	390
3.11 Trinity Aquifer System, Edwards-Trinity Aquifer System	393
3.12 Dougherty Plain and Marianna Lowlands, Floridan Aquifer System	396
3.13 Eastern Flatwoods Southshores, Floridan Aquifer System	399
3.14 Lower Coastal Plain, Floridan Aquifer System	402
3.15 Ocala Uplift, Floridan Aquifer System	405
3.16 Sea Island, Floridan Aquifer System	408
3.17 Tifton Upland, Floridan Aquifer System	412

Style Definition: Comment Text

Style Definition: Strong: Font: Arial, 11 pt, Not Bold

Style Definition: Emphasis: Font: (Default) +Body (Cambria), 14 pt, Bold, Not Italic, English (United States)

Style Definition: No Spacing: Font: 11.5 pt

Formatted: Font: 14 pt

Formatted: Not Superscript/ Subscript

Formatted: Font: Times New Roman, 13 pt, Bold

Formatted: Font: 12 pt

Formatted: Font: +Body (Cambria), Bold

Formatted: Font: +Body (Cambria), Bold

Formatted: Font: +Body (Cambria), Bold

Field Code Changed

Formatted: Font: +Body (Cambria)

Formatted: Font: +Body (Cambria), Bold

Formatted: Font: +Body (Cambria), Bold

Formatted: Font: +Body (Cambria), Bold

Field Code Changed

Formatted: Font: +Body (Cambria)

Formatted: Font: +Body (Cambria), Bold

Formatted: Font: +Body (Cambria), Bold

Formatted: Font: +Body (Cambria), Bold

Field Code Changed

Formatted: Font: +Body (Cambria)

3.18 Vidalia Upland, Floridan Aquifer System.....	416
3.19 Catahoula Area, Gulf Coast Regional Aquifer System.....	419
3.20 Houston-Galveston Area, Gulf Coast Regional Aquifer System.....	422
3.21 Lafayette Area, Gulf Coast Regional Aquifer System.....	425
3.22 Southern Hills, Gulf Coast Regional Aquifer System.....	428
3.23 Central High Plains, High Plains.....	430
3.24 Northern High Plains, High Plains.....	433
3.25 Southern High Plains, High Plains.....	436
3.26 Albuquerque Basin, Middle Rio Grande.....	439
3.27 San Luis Valley, Middle Rio Grande.....	443
3.28 Central Mississippi Embayment, Mississippi Embayment.....	445
3.29 Eastern Mississippi Embayment, Mississippi Embayment.....	448
3.30 Delmarva Peninsula, North Atlantic Coastal Plain.....	451
3.31 Maryland Western Shores, North Atlantic Coastal Plain.....	456
3.32 New Jersey Coastal Plain, North Atlantic Coastal Plain.....	459
3.33 North Carolina and Virginia Coastal Plain, North Atlantic Coastal Plain.....	463
3.34 Powder River Basin, Northern Great Plains.....	467
3.35 Williston Basin, Northern Great Plains.....	470
3.36 Eastern Silurian-Devonian Aquifers, Northern Midwest Aquifer System.....	473
3.37 Mississippian-Silurian-Devonian Carbonates, Northern Midwest Aquifer System.....	476
3.38 Northeast Missouri Carbonates, Northern Midwest Aquifer System.....	482
3.39 Northern Cambrian-Ordovician Aquifers, Northern Midwest Aquifer System.....	486
3.40 Upper Carbonate Aquifer, Northern Midwest Aquifer System.....	488
3.41 Western Cambrian-Ordovician Aquifers, Northern Midwest Aquifer System.....	490
3.42 Mesilla Valley, Rincon-Mesilla Valleys.....	493
3.43 Lower Santa Ynez Valley, Santa Ynez Valley.....	496
3.44 Boise Valley and Homedale Murphy Area, Western Snake River Plain.....	498
3.45 Mountain Home Plateau, Western Snake River Plain.....	500
3.46 Antelope Valley.....	502
3.47 Big Bear Valley.....	504
3.48 Bighorn Basin.....	506
3.49 Black Hills Uplift.....	508
3.50 Black Warrior River Aquifer System (Eutaw and McShan Formations and Tuscaloosa Group).....	511

3.51 Castle Hayne Aquifer..... 514

3.52 Coachella Valley..... 518

3.53 Cuyama Valley..... 520

3.54 Denver Basin..... 522

3.55 Eastern Dakota Aquifer..... 524

3.56 Eureka and Eel River and Mad River Plains..... 527

3.57 Garber-Wellington Aquifer..... 529

3.58 Honey Lake Valley..... 531

3.59 Long Island..... 533

3.60 Los Angeles Basin..... 536

3.61 Michigan Basin..... 539

3.62 Mojave Basin..... 542

3.63 Northern Green River Basin..... 544

3.64 Ozark Plateaus Aquifer System..... 547

3.65 Pearl and Chattahoochee Aquifer System..... 550

3.66 Salinas Valley..... 553

3.67 Salt Lake Valley..... 555

3.68 San Pedro Basin..... 557

3.69 Santa Clara-Calleguas Basin..... 559

3.70 Santa Rosa Valley..... 562

3.71 South Park Basin..... 565

3.72 Tijuana-San Diego Basin..... 568

3.73 Upper Santa Ana Basin..... 571

3.74 Utah Lake Valley..... 573

Supplementary Note 4. Multiple rank correlations to account for interrelationships among our two potential explanatory variables..... 575

Supplementary Note 5. Well depth versus modern groundwater plots for study aquifers..... 147

Supplementary Note 6. Sensitivity analyses of statistical relationship **Rank correlations between groundwater withdrawals and the depth** that modern water reaches after visual inspection of scatterplots of depth versus **below which modern groundwater is scarce if we exclude aquifer systems with shallow depths to confined conditions..... 578**

Supplementary Note 6. Potential explanations for deep modern groundwater 239581

Formatted: Font: +Body (Cambria), Bold

Formatted: Font: +Body (Cambria), Bold

Formatted: Font: +Body (Cambria), Bold

Supplementary Note 7. Modern groundwater prevalence in wells defined by the USGS as tapping a confined aquifer 587

Supplementary Note 8. Locations of hydrogeologic cross sections 590

Formatted: Font: +Body (Cambria)

Supplementary Note 1. Supplementary results

Here, a suite of supplementary maps and plots are provided to complement content in the main text.

Well depth versus. Supplementary Fig. 1. Locations of aquifer systems studied in this project. Aquifer titles correspond to those presented in Supplementary Tables 4 and 6. Shaded areas depict the geologic classification for each of the n=91 study aquifer systems (see legend; see also Supplementary Note 3 for classification steps and primary references reviewed to classify each aquifer system. Aquifer system boundaries derive from study areas delineated in numerous local and regional-scale studies; for further details pertaining to the aquifer delineation process see Supplementary Note 4.

Formatted: Font: 11 pt

n=91 aquifer systems, median depth: 37 m (lower-upper quartile: 14-89 m)

Supplementary Fig. 2. Depths below which modern groundwater is scarce across the United States. Aquifers (n=91) are shaded based on depths below which more than 60% of samples contain minimal (<25%) modern water; among our plots for study aquifers the median depth is 37 meters and the lower-upper quartile range is 14-89 meters. This figure is an identical (but enlarged) version of main text Fig. 1a.

- Formatted: Font: 14 pt, Font color: Text 1
- Formatted: Hearing1
- Formatted: Font: 14 pt
- Formatted: Font: 14 pt
- Formatted: Font: 14 pt

n=85 aquifer systems, median depth: 55 m (lower-upper quartile: 21-105 m)

Supplementary Fig. 3. Depths below which modern groundwater is scarce across the United States. Aquifers (n = 85) are shaded based on the depths below which more than 70% of samples contain minimal (<25%) modern water; among our study aquifers, the median depth is 55 meters and the lower-upper quartile range is 21-105 meters. This figure is an identical (but enlarged) version of main text Fig. 4b.

n=74 aquifer systems, median depth: 72 m (lower-upper quartile: 32-147 m)

Supplementary Fig. 4. Depths below which modern groundwater is scarce across the United States. Aquifers (n = 74) are shaded based on the depths below which more than 80% of samples contain minimal (<25%) modern water; among our study aquifers, the median depth is 72 meters, the lower quartile is 32 meters and the lower-upper quartile range is 32–147 meters. This figure is an identical (but enlarged) version of main text Fig. 1c.

Supplementary Fig. 5. The depths below which modern groundwater is scarce in US aquifers. The shallow depth (top of light blue bar) represents the well depth below which more than 60% of samples contain minimal modern groundwater ("minimal modern groundwater" defined as a sample containing less than 25% modern groundwater). Circles represent well depth below which more than 70% contain minimal modern groundwater. The bottom of the dark blue bar represents the depth below which 80% of samples contain minimal modern groundwater. We plot n=85 aquifers for which we had sufficient tritium data to determine the depth below which more than 70% of samples have minimal modern groundwater.

- Sample may contain more than 25% post-1953 (i.e., “modern”) water
- Sample does not contain more than 25% post-1953 (i.e., “modern”) water

Supplementary Fig. 6. Abundance of post-1953 groundwater in ~15 thousand well water samples across the United States. Each point represents one well for which we estimated the proportion of the pumped groundwater comprised of precipitation that fell more recently than the year 1953 (herein “modern water”). Yellow shaded points represent well waters that may contain more than 25% modern water (i.e., samples with relatively high tritium activities are in yellow; $n=10,089$ points). Conversely, blue shaded points represent well waters that contain less than 25% modern water (i.e., samples with relatively low tritium activities (modern water fraction is less than 25%) are in blue; $n=5,395$ points). The boundaries of all aquifers delineated for this project are outlined in light pink shades (see Supplementary Note 4); only those aquifers with sufficient tritium data to determine the depth below which tritium becomes scarce are included in the main text figures (for the locations of these aquifers see previous supplementary figures). For transparency, we present all of the aquifers that we delineated in this figure as light pink polygons in the background of this figure; each polygon represents one delineated aquifer system. We only studied the $n=91$ aquifer systems with sufficient groundwater tritium data for analyses (see Methods).

Supplementary Fig. 7. The depth below which modern groundwater is scarce tends to be deeper where annual groundwater withdrawals are high. a) Plot of annual groundwater withdrawals and the depth below which modern groundwater is scarce; each point represents one aquifer system. The Spearman rank correlation coefficient (ρ) of the data presented in panel a is $\rho = 0.36$ (Spearman P-value < 0.001). The y-axis values present the depth below which >70% of samples have minimal (<25%) modern groundwater; vertical error bars extend to the depth below which >60% of samples have minimal (<25%) modern groundwater (shallower depth) and the depth below which >80% of samples have minimal (<25%) modern groundwater (deeper depth; see legend in upper left corner of plot). Points are colour coded by generalized aquifer system geologic characteristics (e.g., alluvial basins are upward-pointing triangles; see legend in lower right corner of plot). We only plot the n=85 points representing the aquifer systems for which we had sufficient data to determine the depth below which >70% of samples have minimal (<25%) modern groundwater. **b)** Locations of study aquifer systems; each polygon represents one of our study aquifer systems. Colours of each aquifer system correspond to the generalized geologic conditions, following the identical colour scheme to panel a (see legend in lower right corner of figure).

Supplementary Note 2: Geospatial analyses of explanatory variables

We analysed topographic, land-use and climate datasets in an attempt to better understand correlations between the depth to which modern groundwater penetrates and these potential explanatory variables. For each of our study aquifer systems (i.e., applying the spatial boundaries for each study aquifer as delineated following steps detailed in Supplementary Note 4), we quantified each of the following:

- **Average precipitation divided by potential evapotranspiration (unitless)** (i.e., average of values for all grid cells within the study aquifer's boundaries; these "CGIAR-CSI Global Aridity and Global PET Database" data were downloaded November 3, 2021 from <https://egi.aresi.community/data/global-aridity-and-pet-database/>; further attribution: Zomer RJ, Trabucco A, Bossio DA, van Straaten O, Verchot LV, 2008. Climate Change Mitigation: A Spatial Analysis of Global Land Suitability for Clean Development Mechanism Afforestation and Reforestation. *Agric. Ecosystems and Envir.* 126: 67-80. Zomer RJ, Bossio DA, Trabucco A, Yuanjie L, Gupta DC & Singh VP, 2007. Trees and Water: Smallholder Agroforestry on Irrigated Lands in Northern India. Colombo, Sri Lanka: International Water Management Institute. pp 45. (IWMI Research Report 122))
- **Average topographic slope (percent grade)** (i.e., average of values for all grid cells within the study aquifer's boundaries), determined from the SRTM (https://www.usgs.gov/centers/eros/science/usgs-eros-archive-digital-elevation-shuttle-radar-topography-mission-srtm-1-arc?qt-science_center_objects=0#qt-science_center_objects)
- **Average groundwater pumping estimated for the year 2015 (mm/year)** (County-scale data are from the USGS (Dieter, C.A., Maupin, M.A., Caldwell, R.R., Harris, M.A., Ivahnenko, T.I., Lovelace, J.K., Barber, N.L., and Linsey, K.S., 2018, Estimated use of water in the United States in 2015: U.S. Geological Survey Circular 1441, 65 p., <https://doi.org/10.3133/cir1441>); the county-scale at which these groundwater pumping estimates are provided is imperfect, as many of our aquifer systems span multiple counties or only partly overlap with one (or more) county. To estimate groundwater withdrawals for each aquifer system, we intersected (i.e., identified overlap) between our aquifer system boundaries and counties. Next, we estimated groundwater withdrawals ("Pumping" in following equation) within each study area following: $Pumping = \sum (A_i \times P_i) / \sum (A_i)$, where " A_i " represents the area of county i that overlaps the study aquifer system area, and " P_i " represents the estimated area-normalized groundwater withdrawals within the county (i.e., in mm/year)
- **Median* thickness of "permeable layers above bedrock" (meters)** (quoted text from quote https://daac.ornl.gov/SOILS/guides/Global_Soil_Regolith_Sediment.html; *median calculated by taking the median of the set of values derived from all grid cells within the study aquifer's boundaries), derived from gridded data downloaded November 3, 2021 from https://daac.ornl.gov/SOILS/guides/Global_Soil_Regolith_Sediment.html (see also published work by: Pelletier, J.D., Broxton, P.D., Hazenberg, P., Zeng, X., Troch, P.A., Niu, G.Y., Williams, Z., Brunke, M.A., Goehis, D. (2016). A gridded global data set of soil, intact regolith, and sedimentary deposit thicknesses for regional and global land surface modeling. *Journal of Advances in Modeling Earth Systems*, 8, 41-65.). The maximum value is 50 (i.e., 50 meters); we therefore chose to calculate a median (rather than an average value) because of this truncation in the gridded data product. Aquifer systems where the median value is 50 meters should be interpreted as ≥ 50 meters.

These values, as well as our tritium based estimates of the depth to which modern groundwaters penetrate, are provided in the table below. We provide a plot of modern groundwater versus well depth for each individual aquifer system in Supplementary Figs. 8-98 (see Supplementary Note 5).

Supplementary Table 1. The depth below which * most wells contain minimal (<25%) modern groundwater in our study aquifer systems (columns 3, 4 and 5; see footnote underneath table for quantitative definitions of “most”) and physiographic, water use and climate conditions within the boundaries of each aquifer system (i.e., four potential explanatory variables are in rightmost 4 columns)

Aquifer system title	Broader aquifer system title	* 60% (m below land surf.)	* 70% (m below land surf.)	* 80% (m below land surf.)	Average precipitation divided by potential evapotranspiration (unitless)	Average topographic slope (percent grade)	Average groundwater pumping estimated for the year 2016 (mm/year)	Median thickness of “permeable layers above bedrock” (m)
Sacramento Basin	California Central Valley	92	114	158	0.30	2.2	118	50
San Joaquin Basin	California Central Valley	134	163	203	0.18	1.6	128	50
Tulare Basin	California Central Valley	126	220	232	0.10	1.8	105	50
Central Carrizo-Wilcox	Carrizo-Wilcox	16	16	16	0.57	2.4	10	1
Eastern Carrizo-Wilcox	Carrizo-Wilcox	15	15	20	0.75	3.1	7	1
Western Carrizo-Wilcox	Carrizo-Wilcox	30	30	-	0.32	1.7	12	1
Eagle Valley	Carson River Basin	12	72	152	0.16	12.8	16	1
Central Wabash and Bloomington Ridged Plain	Central Lowland-Till Plain	7	9	10	0.81	1.4	8	50
Blue Mountains and Clearwater Embayment	Columbia Plateau Regional Aquifer System	31	83	-	0.46	19.6	6	1
Palouse Slope	Columbia Plateau Regional Aquifer System	30	30	78	0.33	10.1	15	9
Umatilla Basin and Horse	Columbia Plateau	16	16	62	0.24	10.0	28	2

Aquifer-system title	Broader aquifer system title	*60% (m below land surf.)	*70% (m below land surf.)	*80% (m below land surf.)	Average precipitation divided-by-potential evapotranspiration (unitless)	Average topographic slope (percent grade)	Average groundwater pumping estimated for the year 2015 (mm/year)	Median thickness of "permeable layers above bedrock" (m)
Heaven-Hills	Regional Aquifer System							
Walla-Walla Basin	Columbia Plateau Regional Aquifer System	40	40	153	0.34	13.8	18	8
Yakima-Basin	Columbia Plateau Regional Aquifer System	113	126	130	0.20	9.5	13	4
Stockton-Plateau	Edwards-Trinity Aquifer System	130	-	-	0.14	5.0	4	4
Trinity-Aquifer System	Edwards-Trinity Aquifer System	62	66	73	0.48	2.8	5	4
Bacon-Terrace	Floridan Aquifer System	5	7	42	0.78	1.9	40	50
Dougherty-Plain and Marianna Lowlands	Floridan Aquifer System	76	113	163	0.88	2.7	19	4
Eastern Flatwoods Southshores	Floridan Aquifer System	38	46	64	0.72	1.3	50	50
Intermediate Aquifer	Floridan Aquifer System	47	67	-	0.71	1.5	41	50
Lower-Coastal Plain	Floridan Aquifer System	9	34	59	0.79	2.1	8	50
Ocala-Uplift	Floridan Aquifer System	103	-	-	0.83	2.0	23	40
Sea-Island	Floridan Aquifer System	6	59	75	0.80	1.9	25	50

Aquifer-system title	Broader aquifer system title	*60% (m below land surf.)	*70% (m below land surf.)	*80% (m below land surf.)	Average precipitation divided-by-potential evapotranspiration (unitless)	Average topographic slope (percent grade)	Average groundwater pumping estimated for the year 2015 (mm/year)	Median thickness of "permeable layers above bedrock" (m)
Tifton-Upland	Floridan Aquifer System	9	17	52	0.80	2.6	15	45
Vidalia-Upland	Floridan Aquifer System	7	9	17	0.74	2.6	9	32
Catahoula-Area	Gulf-Coast Regional Aquifer System	73	78	-	1.03	2.9	8	32
Houston-Galveston-Area	Gulf-Coast Regional Aquifer System	6	10	16	0.74	1.8	24	50
Lafayette-Area	Gulf-Coast Regional Aquifer System	3	15	23	0.94	2.0	22	50
Southern-Hills	Gulf-Coast Regional Aquifer System	104	105	178	1.04	2.9	15	46
Central-High Plains	High Plains	35	59	78	0.23	2.1	50	50
Northern-High Plains	High Plains	20	35	64	0.33	3.1	47	32
Southern-High Plains	High Plains	89	99	117	0.20	1.6	69	50
Albuquerque Basin	Middle-Rio Grande	29	87	117	0.13	3.9	18	15
Espanola-Basin	Middle-Rio Grande	24	24	151	0.19	8.7	5	5
San-Luis-Valley	Middle-Rio Grande	107	-	-	0.18	4.6	54	20
Central Mississippi Embayment	Mississippi Embayment	33	37	45	0.87	1.5	153	50
Eastern Mississippi Embayment	Mississippi Embayment	19	29	58	0.95	3.5	11	4

Aquifer-system title	Broader aquifer system title	*-60% (m below land surf.)	*-70% (m below land surf.)	*-80% (m below land surf.)	Average precipitation divided-by-potential evapotranspiration (unitless)	Average topographic slope (percent grade)	Average groundwater pumping estimated for the year 2015 (mm/year)	Median thickness of "permeable layers above bedrock" (m)
Western Mississippi Embayment	Mississippi Embayment	29	29	29	0.90	2.9	11	4
Delmarva Peninsula	North Atlantic Coastal Plain	15	21	29	0.82	1.9	14	50
Maryland Western-Shores	North Atlantic Coastal Plain	8	26	47	0.80	3.6	14	18
New-Jersey Coastal-Plain	North Atlantic Coastal Plain	48	53	58	0.88	1.9	37	50
North-Carolina and-Virginia Coastal-Plain	North Atlantic Coastal Plain	5	8	13	0.85	2.3	11	50
Powder-River Basin	Northern Great Plains	4	16	30	0.24	6.5	2	4
Williston-Basin	Northern Great Plains	7	16	28	0.31	3.5	1	3
Eastern Cambrian-Ordovician Aquifers	Northern Midwest Aquifer System	77	125	-	0.79	5.4	13	12
Eastern Silurian-Devonian Aquifers	Northern Midwest Aquifer System	14	16	29	0.81	2.1	15	48
Mississippian-Silurian-Devonian Carbonates	Northern Midwest Aquifer System	14	32	77	0.75	2.4	10	49
Northeast Missouri Carbonates	Northern Midwest Aquifer System	41	70	-	0.74	3.2	4	37
Northern	Northern	59	73	105	0.79	2.4	16	48

Aquifer-system title	Broader aquifer system title	*-60% (m below land surf.)	*-70% (m below land surf.)	*-80% (m below land surf.)	Average precipitation divided-by-potential evapotranspiration (unitless)	Average topographic slope (percent grade)	Average groundwater pumping estimated for the year 2015 (mm/year)	Median thickness of "permeable layers above bedrock" (m)
Cambrian-Ordovician Aquifers	Midwest Aquifer System							
Upper-Carbonato Aquifer	Northern Midwest Aquifer System	93	134	140	0.74	1.7	4	50
Western Cambrian-Ordovician Aquifers	Northern Midwest Aquifer System	56	87	103	0.77	4.1	13	29
Mesilla-Valley	Rincon-Mesilla Valleys	37	64	93	0.10	3.3	29	18
West-Salt-River Basin	Salt-Basin	45	69	165	0.08	2.2	68	49
Lower-Santa-Ynez-Valley	Santa-Ynez Valley	61	65	71	0.28	11.0	29	4
Valle-de-Juarez-and-Hueco-Bolson	Tularosa-Hueco	119	119	119	0.40	2.6	32	32
Boise-Valley-and-Homedale-Murphy-Area	Western Snake River-Plain	98	191	-	0.18	4.5	107	18
Mountain-Home Plateau	Western Snake River-Plain	127	131	139	0.16	3.3	36	2
Antelope-Valley	-	2	18	84	0.10	6.8	63	11
Big-Bear-Valley	-	122	163	-	0.40	9.6	7	2
Big-Chino-Valley	-	94	120	125	0.16	2.9	4	15
Bighorn-Basin	-	61	61	69	0.19	8.8	3	1
Black-Hills-Uplift	-	398	545	666	0.33	9.2	2	1
Black-Warrior-River-Aquifer-System-(Eutaw-and-McShan-Formations-and-Tuscaloosa-Group)	-	5	15	40	0.95	3.8	5	1

Aquifer system title	Broader aquifer system title	*60% (m below land surf.)	*70% (m below land surf.)	*80% (m below land surf.)	Average precipitation divided by potential evapotranspiration (unitless)	Average topographic slope (percent grade)	Average groundwater pumping estimated for the year 2015 (mm/year)	Median thickness of "permeable layers above bedrock" (m)
Castle Hayne Aquifer	-	24	34	49	0.04	4.6	44	50
Coachella Valley	-	232	277	323	0.05	5.3	24	9
Cuyama Valley	-	67	189	-	0.23	7.9	33	5
Denver Basin	-	34	46	60	0.22	3.4	8	7
Eastern Dakota Aquifer	-	42	24	30	0.63	2.6	5	48
Eureka and Eel River and Mad River Plains	-	59	66	-	4.09	8.6	6	7
Garber-Wellington Aquifer	-	37	49	60	0.54	2.9	40	4
Honey Lake Valley	-	34	75	-	0.22	7.4	8	43
Judith Basin	-	44	44	44	0.32	6.4	4	4
Long Island	-	404	438	449	0.94	2.5	93	46
Los Angeles Basin	-	160	346	397	0.48	2.3	50	50
Michigan Basin	-	89	-	-	0.79	2.4	8	48
Mojave Basin	-	4	34	60	0.06	4.6	8	13
Northern Green River Basin	-	27	49	82	0.20	5.4	2	4
Ozark Plateaus Aquifer System	-	66	113	192	0.78	6.3	7	4
Pearl and Chattahoochee Aquifer System	-	38	55	128	0.82	3.7	14	4
Peedee and Black Creek and Cape Fear Aquifers	-	7	7	44	0.84	4.5	8	50
Salinas Valley	-	89	195	222	0.23	7.8	42	7
Salt Lake Valley	-	294	-	-	0.26	4.5	66	47

Aquifer-system title	Broader aquifer system title	*-60% (m below land surf.)	*-70% (m below land surf.)	*-80% (m below land surf.)	Average precipitation divided-by-potential evapotranspiration (unitless)	Average topographic slope (percent grade)	Average groundwater pumping estimated for the year 2016 (mm/year)	Median thickness of "permeable layers above bedrock" (m)
San-Antonio Creek-Valley	-	24	24	27	0.28	40.6	29	5
San-Pedro-Basin	-	204	244	280	0.48	8.0	24	40
Santa-Clara-Calleguas-Basin	-	77	102	226	0.21	6.7	44	8
Santa-Rosa Valley	-	34	70	209	0.56	4.5	17	19
South-Park-Basin	-	5	12	67	0.26	8.2	0	4
Southern-San Juan-Basin	-	23	23	23	0.17	7.0	4	4
Spanish-Springs Valley	-	5	5	14	0.12	8.8	3	2
Tijuana-San Diego	-	8	8	37	0.16	5.0	13	40
Upper-Santa-Ana Basin	-	330	348	351	0.17	7.5	15	9
Utah-Lake-Valley	-	276	-	-	0.25	5.3	36	13

* the depth below which most samples contain minimal (<25%) modern water, where the value defining "most" is reported in the column headings (columns 3, 4 or 5) as one of (i) 60%, (ii) 70%, or (iii) 80%

Formatted: Font: Not Bold

Supplementary Table 2. Spearman rank correlation coefficients (ρ) describing statistical relationship between ‘the depth below which most* samples contain minimal (<25%) modern water and four potential explanatory variables. Statistically significant (P-value < 0.001) correlation coefficients are in bold

Calculated depth below which most* samples contain minimal (<25%) modern water	Average precipitation divided by potential evapotranspiration (unitless)	Average topographic slope (percent grade)	Average groundwater pumping estimated for the year 2015 (mm/year)	Median thickness of ‘permeable layers above bedrock (m)
most* = 60%	$\rho = -0.24$	$\rho = 0.10$	$\rho = 0.35$ (P < 0.001)	$\rho = -0.05$
most* = 70%	$\rho = -0.28$	$\rho = 0.14$	$\rho = 0.36$ (P < 0.001)	$\rho = -0.02$
most* = 80%	$\rho = -0.39$ (P < 0.001)	$\rho = 0.24$	$\rho = 0.40$ (P < 0.001)	$\rho = -0.09$

* ‘most’ interpreted as 60% (row 2 of table), 70% (row 3 of table) or 80% (row 4 of table)

Formatted: Font: 10 pt

Supplementary Table 3. Multiple regression of the rank transforms coefficients (β) describing statistical relationship between ‘the depth below which most* samples contain minimal (<25%) modern water and four potential explanatory variables. Statistically significant (P value < 0.001) partial regression coefficients (β) are in bold

Calculated depth below which most* samples contain minimal (<25%) modern water	Average precipitation divided by potential evapotranspiration (unitless)	Average topographic slope (percent grade)	Average groundwater pumping estimated for the year 2015 (mm/year)	Median thickness of ‘permeable layers above bedrock (m)
most* = 60%	$\beta = -0.13$ (P > 0.05)	$\beta = 0.01$ (P > 0.05)	$\beta = 0.41$ (P < 0.001)	$\beta = -0.18$ (P > 0.05)
most* = 70%	$\beta = -0.15$ (P > 0.05)	$\beta = 0.19$ (P > 0.05)	$\beta = 0.40$ (P < 0.001)	$\beta = -0.01$ (P > 0.05)
most* = 80%	$\beta = -0.23$ (P > 0.05)	$\beta = 0.23$ (P > 0.05)	$\beta = 0.45$ (P < 0.001)	$\beta = -0.04$ (P > 0.05)

* ‘most’ interpreted as 60% (row 2 of table), 70% (row 3 of table) or 80% (row 4 of table)

Formatted: Font: 10 pt

Supplementary Table 4. Spearman rank correlation coefficients (ρ) describing statistical relationship between ‘the depth below which most* samples contain minimal (<25%) modern water and annual groundwater withdrawals, when our regression analyses are separated into two categories of aquifers: (i) alluvial basins (see triangles in Fig. 3) and (ii) non alluvial basins.

	Aquifer systems with geologic categories that ‘are not’ categorized as alluvial basins: Rank correlations between ‘Calculated depth below which most* samples contain minimal (<25%) modern water’ versus ‘Average groundwater pumping estimated for the year 2015 (mm/year)’	Aquifer systems with geologic categories that ‘are’ categorized as alluvial basins: Rank correlations between ‘Calculated depth below which most* samples contain minimal (<25%) modern water’ versus ‘Average groundwater pumping estimated for the year 2015 (mm/year)’
most* = 60%	$\rho = 0.17$ (P = 0.18; N=60)	$\rho = 0.38$ (P = 0.03; N=31)
most* = 70%	$\rho = 0.22$ (P = 0.10; N=57)	$\rho = 0.33$ (P = 0.08; N=28)
most* = 80%	$\rho = 0.23$ (P = 0.10; N=51)	$\rho = 0.34$ (P = 0.11; N=23)

* ‘most’ interpreted as 60% (row 2 of table), 70% (row 3 of table) or 80% (row 4 of table)

Formatted: Font: 10 pt

Supplementary Note 3. Classified geologic typologies for study aquifers. The following Supplementary Figs. 1-91

We assessed geologic conditions in our aquifers for which we quantified a depth at which modern groundwater becomes scarce (see main text figures). To do so, we reviewed local and regional scale hydrogeologic studies for all of the study aquifers for which we could identify a depth below which modern groundwater is scarce (see Methods) in the aquifer system; aquifer system boundaries themselves were delineated following local and regional scale studies (see Supplementary Note 4). We classified our n=91 study aquifer systems into one of the 11 categories in Supplementary Table 5. The classifications for each aquifer system are presented in Supplementary Table 6, along with primary references and quoted text from these cited works that form the basis for our aquifer system classification. The median and lower-upper quartile range of depths below which modern groundwater becomes scarce (main text methods) determined on the basis of study areas with a given classification are in Supplementary Note 4.

Formatted: Font: Times New Roman, English (United Kingdom)

Supplementary Table 5. Eleven aquifer system classifications based on geologic conditions

Classification (generalized)	General characteristics
Thick alluvium and poorly/semi-consolidated sedimentary formations	Alluvial basins surrounded by steep topography (e.g., California's Central Valley)
Poorly/semi-consolidated dipping layered clastic sedimentary sequences	Clastic sedimentary deposits that are unconsolidated or weakly consolidated, often with relatively thin alluvium (e.g., the eastern portion of the Mississippi Embayment Aquifer System where the Mississippi River Valley Alluvial Aquifer is absent)
Consolidated clastic sedimentary rock formations	Consolidated clastic sedimentary rock formations (e.g., sandstones, mudstones; e.g., Garber-Wellington Aquifer in Central Oklahoma)
Moderately thick unconsolidated aquifer overlying layered clastic sedimentary sequences	Aquifer systems with a shallow (near surface) unconsolidated alluvial aquifer of thin or moderate thickness (less than ~50 m in areas) that is underlain by layered sedimentary sequences of varying permeability (e.g., the central portion of the Mississippi Embayment where the Mississippi River Valley Alluvial Aquifer is present)
Glacial drift overlying consolidated clastic sedimentary sequences	Glacial drift deposited during the Pleistocene overlies clastic sedimentary formations (e.g., the Central Lowland Till Plain)
Glacial drift overlying poorly/semi-consolidated dipping layered clastic sedimentary sequences	Glacial drift deposited during the Pleistocene overlies poorly-consolidated sedimentary formations (e.g., Long Island)
Carbonate rocks interbedded with clastic sedimentary rocks	Carbonate rocks (e.g., limestone, dolostone) are common; clastic sedimentary rock layers are present and separate carbonate layers (e.g., Ozark Plateaus Aquifer System)
Unconsolidated deposits overlying carbonate rocks	Unconsolidated deposits (not of glacial origin) overlie carbonate rocks (which may be interbedded with consolidated clastic rocks; e.g., the Sea Island physiographic province in the eastern portion of the Floridan Aquifer System)
Glacial drift overlying carbonate rocks	Glacial drift deposited during the Pleistocene overlies carbonate rock formations (e.g., the Silurian-Devonian Aquifers in the Northern Midwest Aquifer System)
Volcanic (basalt) rocks and alluvial deposits	Aquifer system characterized by basalt flow deposits and alluvium (e.g., Columbia Plateau Aquifer System)
Consolidated sedimentary rocks (clastic and carbonate)	Consolidated sedimentary formations including clastic and chemical (carbonate) rock types, where neither rock type dominates the area (e.g., South Park Basin in Colorado)

Supplementary Table 6. Geologic typology classifications for study aquifer systems

Aquifer system title	Broader aquifer system title	Classification*	Basis for classification (cited figures are geologic cross-sections in most cases – text often quoted directly from reference in adjacent column (i.e., the column to the right))	Reference(s)
Sacramento Basin	California Central Valley	Thick alluvium and poorly/semi-consolidated sedimentary formations	"The Tehama Formation is a 600-900 m thick fluvial deposit that extends from the Coast Ranges to the axis of the Sacramento Valley," and "...In the subsurface the Tehama deposits generally are water-saturated and unconsolidated, except near the base of the section where moderately consolidated gravels occur..." Text is quoted directly from Davisson and Criss (1993).	Davisson, M. L., Criss, R. E. (1993). Stable isotope imaging of a dynamic groundwater system in the southwestern Sacramento Valley, California, USA. Journal of Hydrology , 144, 243-246.
San Joaquin Basin	California Central Valley	Thick alluvium and poorly/semi-consolidated sedimentary formations	Fig. 3 of Myer (2005); "The basin is filled with predominantly marine Cretaceous and Cenozoic elastic sedimentary rocks that attain an aggregate thickness of over 9,150 m (30,000 ft)." Text is quoted directly from Myer et al. (2005).	Myer, L., Downey, C., Clinkenbeard, J., Thomas, S., Stevens, S., Benson, S., Zheng, H., Herzog, H., Biediger, B. (2005). Preliminary Geologic Characterization of West Coast States for Geologic Sequestration. West Coast Regional Carbon Sequestration Partnership Report, DOE Contract No. DE-FC26-03NT41984, 53 pp. Accessed November 28, 2021 from https://doi.org/10.2472/907916
Tulare Basin	California Central Valley	Thick alluvium and poorly/semi-consolidated sedimentary formations	Fig. 3 of Page (1983); "The Tulare Formation and other continental deposits of Pliocene to Holocene age crop out over most of the 1,040 square mile area near Kettleman City in the San Joaquin Valley of California. These deposits range in thickness from 0 to more than 4,000 feet and overlie the upper Mya zone of the San Joaquin Formation of Pliocene age." and "Sediments in the Tulare Formation and other continental deposits consist mainly of unconsolidated deposits of clay, silt, sand, and gravel." Text quoted directly from Page (1983).	Page, R. W. (1983). Geology of the Tulare Formation and other continental deposits, Kettleman City area, San Joaquin Valley, California, with a section on ground-water management considerations and use of texture maps (No. 83-4000). U.S. Geological Survey Water Resources Investigations Report 83-4000, 28 pp. Accessed November 28, 2021 from https://pubs.usgs.gov/wri/1983/4000/report.pdf
Central Carrizo-Wileox	Carrizo-Wileox	Poorly/semi-consolidated dipping layered elastic sedimentary sequences	Fig. 2.4 of George (2009); "The Carrizo-Wileox Aquifer is located within the Interior Coastal Plains sub-province of the Gulf Coastal Plains physiographic province (Wermund, 1996). The Interior Coastal Plains are characterized by alternating sequences of unconsolidated sands and clays."	George, P.G. (2009). Geology of the Carrizo-Wileox Aquifer. In: Aquifers of the upper coastal plains of Texas (eds. Hutchison, W.R., Davidson, S.C., Brown, B.J., Mace, R.E.), Chapter 2, Texas Water Development Board Report 374, pp. 17-34. Accessed May 19, 2021 from https://www.twdb.texas.gov/publications/reports/numbered_report/s/doc/R374_AquifersofUpperCoastalPlains.pdf
Eastern Carrizo-Wileox	Carrizo-Wileox	Poorly/semi-consolidated dipping layered elastic sedimentary sequences	Fig. 2.4 of George (2009); "The Carrizo-Wileox Aquifer is located within the Interior Coastal Plains sub-province of the Gulf Coastal Plains physiographic province (Wermund, 1996). The Interior Coastal Plains are characterized by alternating sequences of unconsolidated sands and clays."	George, P.G. (2009). Geology of the Carrizo-Wileox Aquifer. In: Aquifers of the upper coastal plains of Texas (eds. Hutchison, W.R., Davidson, S.C., Brown, B.J., Mace, R.E.), Chapter 2, Texas Water Development Board Report 374, pp. 17-34. Accessed May 19, 2021 from https://www.twdb.texas.gov/publications/reports/numbered_report/s/doc/R374_AquifersofUpperCoastalPlains.pdf

Formatted: Font: Arial, 10 pt,

Aquifer system title	Broader aquifer system title	Classification*	Basis for classification (cited figures are geologic cross-sections in most cases—text often quoted directly from reference in adjacent column (i.e., the column to the right))	Reference(s)
Western Carrizo-Wilcox	Carrizo-Wilcox	Poorly/semi-consolidated dipping layered-elastic sedimentary sequences	Fig. 2.4 of George (2009); "The Carrizo-Wilcox Aquifer is located within the Interior Coastal Plains sub-province of the Gulf Coastal Plains physiographic province (Wernumund, 1996). The Interior Coastal Plains are characterized by alternating sequences of unconsolidated sands and clays."	George, P.G. (2009). Geology of the Carrizo-Wilcox Aquifer. In: Aquifers of the upper coastal plains of Texas (eds. Hutchison, W.R., Davidson, S.C., Brown, B.J., Mace, R.E.), Chapter 2, Texas Water Development Board Report 374, pp. 17-34. Accessed May 19, 2021 from https://www.twdb.texas.gov/publications/reports/numbered_reports/doc/R374_AquifersofUpperCoastalPlains.pdf
Eagle Valley	Carson River Basin	Thick alluvium and poorly/semi-consolidated sedimentary formations	"Basin-fill sediments that overlie bedrock are generally coarse grained near the base of the mountains and finer grained near the center of the valley. These sediments form the principal groundwater reservoir for municipal supply and form the basin-fill aquifer. They are estimated to be about 1,200 ft thick 1.5 mi west of Lone Mountain, about 400 to 800 ft thick beneath the northeastern and southern parts of the valley, and about 2,000 ft thick about 1 mi northwest of Prison Hill (Artega, 1986, p. 25)." Text quoted directly from Maurer and Thodal (2000).	Maurer, D.K., Thodal, C.E. (2000). Quantity and chemical quality of recharge, and updated water budgets, for the basin-fill aquifer in Eagle Valley, western Nevada. U.S. Geological Survey Water Resources Investigations Report 99-4289, 52 pp. Accessed November 29, 2021 from https://pubs.usgs.gov/wri/1999/4289/report.pdf
Central Wabash and Bloomington Ridged Plain	Central Lowland Till Plain	Glacial drift overlying consolidated-elastic sedimentary sequences "glacial drift common in northern portion of area	Cross sections depicted in figures on pages 18 and 19 by Stephenson (1967). "The study area lies primarily within the Bloomington Ridged Plain of the Till Plain Section, Central Lowland Province (Leighton, Ekblaw, and Horberg, 1948, p. 18). The Bloomington Ridged Plain has wide stretches of relatively flat or gentle undulatory surfaces composed of ground moraine punctuated by a series of low, broad morainic ridges" and "Unconsolidated deposits of Pleistocene and Recent ages overlie the eroded bedrock surface in all of the study area and reach a maximum thickness of over 400 feet in some portions of bedrock channels." and "The major feature of the bedrock map is the Mahomet Valley, which takes its name from the village of Mahomet located over the deepest part of the channel in Champaign County (fig. 6). The bottom elevation of the valley averages 350 feet, placing it 200 to 300 feet below adjoining uplands" and "The upland bedrock surface is developed primarily on relatively weak shale." Text quoted directly from Stephenson (1967).	Stephenson, D.A. (1967). Hydrogeology of glacial deposits of the Mahomet Bedrock Valley in east-central Illinois. Illinois State Geological Survey Circular 409, 56 pp. Accessed November 29, 2021 from https://www.ideals.illinois.edu/bitstream/handle/2142/35125/hydrogeologyofgl409step.pdf?sequence=2
Blue Mountains and Clearwater Embayment	Columbia Plateau Regional Aquifer System	Volcanic (basalt) rocks and alluvial deposits	"The CPRAS includes seven hydrogeologic units—the Overburden unit, three units in the permeable basalt rock, two confining units, and the basement confining unit. The three basalt units are the Saddle Mountains, Wanapum, and Grande Ronde Basalts and their intercalated sediments. Median thickness of the Saddle Mountains and Wanapum units are 280 and 330 feet, respectively. Median thickness of the deepest and most-	Kahle, S.C., Morgan, D.S., Welch, W.B., Ely, S.R., Vaccaro, J.J., Orzol, L.L. (2011). Hydrogeologic Framework and Hydrologic Budget Components of the Columbia Plateau Regional Aquifer System, Washington, Oregon, and Idaho. U.S. Geological Survey Scientific Investigations Report 2011-5124, 80 pp. Accessed February 16, 2021 from https://pubs.usgs.gov/sir/2011/5124/pdf/sir20115124.pdf

Aquifer system title	Broader aquifer system title	Classification*	Basis for classification (cited figures are geologic cross-sections in most cases—text often quoted directly from reference in adjacent column (i.e., the column to the right))	Reference(s)
			voluminous unit, the Grande Ronde unit, is largely unknown, but total thickness may exceed 14,000 feet near the center of the basin. The confining units are equivalent to the Saddle Mountains-Wanapum and Wanapum-Grande Ronde interbeds, referred to in this study as the Mabton and Vantage Interbeds, respectively. The interbed units are fairly extensive laterally, but are thin (generally tens of feet), especially when compared with the thickness of the basalt units. The basement confining unit, referred to as Older Bedrock, consists of pre-CRBG rocks that generally have much lower permeabilities than the basalts and are considered the base of the regional flow system” Text quoted directly from Kahle et al. (2011).	
Palouse Slope	Columbia Plateau Regional Aquifer System	Volcanic (basalt) rocks and alluvial deposits	“The CPRAS includes seven hydrogeologic units—the Overburden unit, three units in the permeable basalt rock, two confining units, and the basement confining unit. The three basalt units are the Saddle Mountains, Wanapum, and Grande Ronde Basalts and their intercalated sediments. Median thickness of the Saddle Mountains and Wanapum units are 280 and 330 feet, respectively. Median thickness of the deepest and most voluminous unit, the Grande Ronde unit, is largely unknown, but total thickness may exceed 14,000 feet near the center of the basin. The confining units are equivalent to the Saddle Mountains-Wanapum and Wanapum-Grande Ronde interbeds, referred to in this study as the Mabton and Vantage Interbeds, respectively. The interbed units are fairly extensive laterally, but are thin (generally tens of feet), especially when compared with the thickness of the basalt units. The basement confining unit, referred to as Older Bedrock, consists of pre-CRBG rocks that generally have much lower permeabilities than the basalts and are considered the base of the regional flow system” Text quoted directly from Kahle et al. (2011).	Kahle, S.C., Morgan, D.S., Welch, W.B., Ely, S.R., Vaccaro, J.J., Orzol, L.L. (2011). Hydrogeologic Framework and Hydrologic Budget Components of the Columbia Plateau Regional Aquifer System, Washington, Oregon, and Idaho. US Geological Survey Scientific Investigations Report 2011-5124, 80 pp. Accessed February 16, 2021 from https://pubs.usgs.gov/sir/2011/5124/pdf/sir20115124.pdf
Umatilla Basin and Horse Heaven Hills	Columbia Plateau Regional Aquifer System	Volcanic (basalt) rocks and alluvial deposits	Fig. 4 of Herrera (2017): “The upper Umatilla River Basin lies within the Columbia River flood-basalt province and is underlain by a thick sequence of Miocene-age volcanic flood-basalts that collectively make up the Columbia River Basalt Group (CRBG) (figs. 3 and 4A–B). In parts of the upper Umatilla River Basin, the CRBG lava flows are overlain by Miocene to Holocene sedimentary deposits of conglomerate, loess, silt, sand, and recent alluvium. The CRBG lava flows are underlain by Eocene to Oligocene volcanic rocks, and Paleocene to Eocene arkosic sandstones assigned to the Herren Formation (generally included as part of the Clarno Formation), and pre-Tertiary amphibolite	Herrera, N. B., Ely, K., Mehta, S., Stonewall, A. J., Risley, J. C., Hinkle, S. R., Conlon, T. D. (2017). Hydrogeologic framework and selected components of the groundwater budget for the upper Umatilla River Basin, Oregon (No. 2017-5020). US Geological Survey Scientific Investigations Report 2017-5020, 68 pp. Accessed February 18, 2021 from https://pubs.usgs.gov/sir/2017/5020/sir20175020.pdf Davies-Smith, A., Bolke, E. L., Collins, C. A. (1988). Geohydrology and digital simulations of the ground-water flow system in the Umatilla Plateau and Horse Heaven Hills

Aquifer system title	Broader aquifer system title	Classification*	Basis for classification (cited figures are geologic cross-sections in most cases—text often quoted directly from reference in adjacent column (i.e., the column to the right))	Reference(s)
			schist and gneiss, intrusive norite and quartz diorite." "The main avenues of ground-water movement in the Columbia River Basalt Group are the interflow zones between basalt layers. These complex, poorly interconnected zones may be rather extensive in the lateral direction but largely isolated from overlying and underlying interflows by poorly permeable basalt flow centers. Wells in the Columbia River Basalt Group usually intercept more than one interflow zone. Another control on ground-water movement is the "barrier" effect of stratigraphic pinch-outs and offsets. Although these are not entirely impermeable to ground-water, they do retard ground-water movement and may effectively isolate parts of the units." Text quoted from Davies-Smith et al. (1988). "Geohydrologic units (aquifers) in the study area were delineated by vertical grouping of wells with similar head, separated by zones of low permeability. Four such units were defined for this report. The uppermost unit (layer 1) consists of unconsolidated deposits of gravel, sand, silt, and clay that overlie the basalt; layer 2 consists of the Saddle Mountains Basalt, layer 3 the Wanapum Basalt, and layer 4 consists of the upper thousand feet of Grande Ronde Basalt." Text quoted from Davies-Smith et al. (1988)	area, Oregon, and Washington: U.S. Geological Survey Water Resources Investigations Report 87-4268, 77 p. https://pubs.usgs.gov/wri/1987/4268/report.pdf
Walla-Walla Basin	Columbia Plateau Regional Aquifer System	Volcanic (basalt) rocks and alluvial deposits	Fig. 6 of Vaccaro et al. (2015) "The only consolidated rock exposed in the Walla-Walla Basin is the Columbia River Basalt, a thick sequence of lava flows of generally basaltic composition." Text quoted from Newcomb (1965). "The modeled area shown in figure 1 is underlain by about 3,000 feet of layered basalt of the Columbia River Group. This basalt sequence is a thick accumulation of individual lava flows that taper and overlap laterally. Individual flows may reach a maximum of 150 feet in thickness. Contact (interflow) zones between the successive flows, resulting from interruptions of flow during the outpouring of the basaltic lavas, comprise about 10 percent of a typical section in the basalt (Newcomb, 1965, p. 29). Because contact zones between individual lava flows are generally rubble, containing angular, broken material, they provide conduits through which ground water may be readily transmitted.	Vaccaro, J.J., Kahle, S.C., Ely, D.M., Burns, E.R., Snyder, D.T., Haynes, J.V., Olsen, T.D., Welch, W.B., and Morgan, D.S. (2015). Groundwater availability of the Columbia Plateau Regional Aquifer System, Washington, Oregon, and Idaho: U.S. Geological Survey Professional Paper 1817, 87 pp. Accessed November 29, 2021 from https://pubs.usgs.gov/pp/1817/pp1817.pdf Newcomb, R.C. (1965). Geology and ground-water resources of the Walla-Walla River Basin, Washington-Oregon: Washington Division of Water Resources Water Supply Bulletin 21, 162 pp. Accessed November 29, 2021 from https://apps.ecology.wa.gov/publications/documents/wsb21.pdf MacNish, R.D., and Barker, R.A. (1976). Digital simulation of a basalt aquifer system, Walla-Walla River basin, Washington and Oregon: State of Washington, Department of Ecology, Water Supply Bulletin 44. Accessed from: https://apps.ecology.wa.gov/publications/documents/wsb44.pdf

Aquifer system title	Broader aquifer system title	Classification*	Basis for classification (cited figures are geologic cross-sections in most cases — text often quoted directly from reference in adjacent column (i.e., the column to the right))	Reference(s)
			In contrast to the horizontal permeable zones, many lava flows, especially the thinner flows, are traversed vertically by narrow cracks that developed as the lava contracted during cooling. Water transmission through these joints occurs very slowly in comparison to flow through the horizontal conduit zones in the basalt. Thus, the occurrence and movement of water in the basalt is greatly influenced by the physical nature of the flows and associated inter-zones." Text quoted from MacNish and Barker (1976). "The hydraulic conductivity of horizontal permeable zones may be as large as 5 ft/s, while the vertical hydraulic conductivity may be as small as 0.000000005 ft/sec. In other words, the rates of horizontal flow in the basalt may be as much as a billion times greater than for vertical flow through the basalt. In places the sequence of basalt flows has been disrupted by movements of the earth's crust which occurred after the flows were laid down. Some of these disruptions are exemplified by the bending and warping of flows and interflow zones. In other places more intense disruptions have produced cracks that may traverse vertically through several flows and extend thousands of feet horizontally. When such folds "pinch off" porous interflow zones, or earth movement along the cracks offsets the interflow zones or plugs them with finely pulverized rock, these blockages can significantly reduce the rates of transmission of ground water." Text quoted from MacNish and Barker (1976).	
Yakima Basin	Columbia Plateau Regional Aquifer System	Volcanic (basalt) rocks and alluvial deposits	Fig. 7b of Kahle et al. (2011): "Within the Yakima Fold Belt (fig. 1), Miocene sedimentary deposits of the Ellensburg Formation underlie, intercalate, and overlie the CRBG and comprise most of the thickness of the unconsolidated deposits in the basinal areas (Jones and others, 2006). These continental sedimentary deposits include fluvial sands and gravels, overbank deposits, lacustrine deposits, alluvial fan deposits, sandstone, conglomerate, and interbedded volcanoclastic sediments." Text quoted directly from Kahle et al. (2011).	Kahle, S.C., Morgan, D.S., Welch, W.B., Ely, S.R., Vaccaro, J.J., Orzol, L.L. (2011). Hydrogeologic Framework and Hydrologic Budget Components of the Columbia Plateau Regional Aquifer System, Washington, Oregon, and Idaho. US Geological Survey Scientific Investigations Report 2011-5124, 80 pp. Accessed February 16, 2021 from https://pubs.usgs.gov/sir/2011/5124/pdf/sir20115124.pdf
Stockton Plateau	Edwards-Trinity Aquifer System	Consolidated sedimentary rocks (clastic and carbonate)	Fig. 6-13 by Bruun et al. (2016): "The Edwards-Trinity (Plateau) Aquifer is a major aquifer extending across much of the southwestern part of the state (Figure 6-11). The water-bearing units are composed predominantly of limestone and dolomite of the Edwards Group and sands of the Trinity Group." text quoted from Bruun et al. (2016)	Bruun, B., Jackson, K., Lake, P., Walker, J. (2016). Texas Aquifers Study. Texas Water Development Board Report 336 pp. Accessed April 1, 2021 from https://www.twdb.texas.gov/groundwater/docs/studies/TexasAquifersStudy_2016.pdf#page=100

Aquifer system title	Broader aquifer system title	Classification*	Basis for classification (cited figures are geologic cross-sections in most cases—text often quoted directly from reference in adjacent column (i.e., the column to the right))	Reference(s)
			"In the Trans-Pecos area (the area west of the Pecos River) and Edwards Plateau area of western and west-central Texas, the Edwards-Trinity aquifer consists of rocks of the Washita, the Frederickeburg, and the Trinity Stages, and the Coahuilan Series" and "The rocks that compose the Edwards-Trinity aquifer are generally limestone in the upper part and sand and sandstone in the lower part." text quoted directly from Ryder (1996) "The southern part of the Trans-Pecos, the Stockton Plateau, has more rugged terrain of exposed carbonate rocks lacking any alluvial mantle." (Kuniansky and Ardis, 2004; page 7)	Ryder, P. (1996). Ground Water Atlas of the United States Segment 4 Oklahoma, Texas. Hydrologic Investigations Atlas 730-E, 32 pp. Accessed November 20, 2021 from https://pubs.usgs.gov/ha/730e/report.pdf Kuniansky, E. L., & Ardis, A. F. (2004). Hydrogeology and ground-water flow in the Edwards-Trinity aquifer system, west-central Texas. US Department of the Interior, US Geological Survey. https://pubs.usgs.gov/pp/1421a/report.pdf
Trinity Aquifer System	Edwards-Trinity Aquifer System	Consolidated elastic sedimentary rock formations	"The Trinity aquifer consists of sandstone, sand, silt, clay, conglomerate, shale, limestone, dolomite, and marl of the Trinity Stage and the Coahuilan Series." Text quoted from Ryder (1996). In these areas, the Paluxy and Travis Peak Formations or the Paluxy and Twin Mountains Formations coalesce to form an undifferentiated unit mostly of sand and sandstone that is referred to as the "Antlers Formation" in many reports. (Ryder (1996), page E24)	Ryder, P. (1996). Ground Water Atlas of the United States Segment 4 Oklahoma, Texas. Hydrologic Investigations Atlas 730-E, 32 pp. Accessed November 20, 2021 from https://pubs.usgs.gov/ha/730e/report.pdf
Bacon Terrace	Floridan Aquifer System	Unconsolidated deposits overlying carbonate rocks *surficial deposits are up to ~30 m thick see Fig. 20 by Williams and Kuniansky (2016).	"A thick sequence of carbonate rocks (limestone and dolomite) of Tertiary age comprise the Floridan aquifer system." Text quoted from Miller (1990). *Bacon Terrace is within The Southeastern Coastal Plain aquifer system (Miller, 1990, page 18). "predominantly warm, shallow marine, platform carbonate rocks that have been deposited in a thick continuous sequence beneath southeastern Georgia and the Florida peninsula" (Williams and Kuniansky, 2015, page 10)	Miller, J.A. (1990). Ground Water Atlas of the United States: Segment 6, Alabama, Florida, Georgia, South Carolina. U.S. Geological Survey Hydrologic Atlas 730-G, 30 pp. Accessed April 5, 2021 from https://www.nre.gov/docs/ML1706/ML17060B027.pdf Williams, L. J., & Kuniansky, E. L. (2015). Revised hydrogeologic framework of the Floridan aquifer system in Florida and parts of Georgia, Alabama, and South Carolina. US Department of the Interior, US Geological Survey.
Dougherty Plain and Marianna Lowlands	Floridan Aquifer System	Carbonate rocks interbedded with elastic sedimentary rocks *surficial deposits are thin see Fig. 20 by Williams and Kuniansky (2016)	"A thick sequence of carbonate rocks (limestone and dolomite) of Tertiary age comprise the Floridan aquifer system." Text quoted from Miller (1990). "The surficial geology of the Dougherty Plain consists of a residual layer of sand and clay, derived from solution weathering of the Ocala Limestone." (Watson, 1981) "The capacity of the principal artesian aquifer to store and transmit large quantities of water is due largely to the cavernous nature of	Miller, J.A. (1990). Ground Water Atlas of the United States: Segment 6, Alabama, Florida, Georgia, South Carolina. U.S. Geological Survey Hydrologic Atlas 730-G, 30 pp. Accessed April 5, 2021 from https://www.nre.gov/docs/ML1706/ML17060B027.pdf Williams, L.J., Kuniansky, E.L. (2016). Revised hydrogeologic framework of the Floridan aquifer system in Florida and parts of Georgia, Alabama, and South Carolina. U.S. Geological Survey Professional Paper 1807, 140 pp. Accessed March 31, 2021 from https://pubs.usgs.gov/pp/1807/pdf/pp1807.pdf

Aquifer system title	Broader aquifer system title	Classification*	Basis for classification (cited figures are geologic cross-sections in most cases—text often quoted directly from reference in adjacent column (i.e., the column to the right))	Reference(s)
			the Ocala Limestone. Water moving through small fractures or cracks in the limestone has slowly enlarged these features, through solution, forming a cavernous, highly porous labyrinth of subterranean channels. In many areas, ground water moves through the aquifer almost as if in a conduit or culvert." Text quoted from Watson (1981). "...limestone is thinly covered with surficial materials or weathered into a residuum." Text quoted from Williams and Kuniansky (2016).	Watson, T. W. (1981). Geohydrology of the Dougherty Plain and adjacent area, southwest Georgia: Georgia Geologic Survey Hydrologic Atlas 5. Accessed from: https://epd.georgia.gov/document/publication/ha-5-geohydrology-dougherty-plain-and-adjacent-area-southwest-georgia-1981/download
Eastern Flatwoods Southshores	Floridan-Aquifer System	Unconsolidated deposits overlying carbonate rocks *surficial deposits are up to ~60 m thick see Fig. 20 by Williams and Kuniansky (2016).	"A thick sequence of carbonate rocks (limestone and dolomite) of Tertiary age comprise the Floridan aquifer system." Text quoted from Miller (1990). "The study area is underlain by a thick sequence of Cretaceous to Holocene unconsolidated and semiconsolidated layers of sand and clay, and poorly indurated to very dense layers of limestone and dolomite." Text quoted from Williams and Kuniansky (2016)	Miller, J.A. (1990). Ground Water Atlas of the United States: Segment 6, Alabama, Florida, Georgia, South Carolina. U.S. Geological Survey Hydrologic Atlas 730-G, 30 pp. Accessed April 5, 2021 from https://www.nrc.gov/docs/ML1706/ML1706B027.pdf Williams, L.J., Kuniansky, E.L. (2016). Revised hydrogeologic framework of the Floridan aquifer system in Florida and parts of Georgia, Alabama, and South Carolina. U.S. Geological Survey Professional Paper 1807, 140 pp. Accessed March 31, 2021 from https://pubs.usgs.gov/pp/1807/pdf/pp1807.pdf
Intermediate Aquifer	Floridan-Aquifer System	Unconsolidated deposits overlying carbonate rocks *surficial deposits are up to ~10 m thick see Fig. 20 by Williams and Kuniansky (2016).	"A thick sequence of carbonate rocks (limestone and dolomite) of Tertiary age comprise the Floridan aquifer system." Text quoted from Miller (1990). "In general, the aquifer system is composed of a complex assemblage of carbonate and siliciclastic sediments with abrupt contacts between facies, resulting in permeable zones that are only locally hydraulically connected. The permeable zones consist of indurated limestone and dolomite, and in some places, unconsolidated elastic material (Knechenmus, 2006)." Text quoted from Williams and Kuniansky (2016).	Miller, J.A. (1990). Ground Water Atlas of the United States: Segment 6, Alabama, Florida, Georgia, South Carolina. U.S. Geological Survey Hydrologic Atlas 730-G, 30 pp. Accessed April 5, 2021 from https://www.nrc.gov/docs/ML1706/ML1706B027.pdf Williams, L.J., Kuniansky, E.L. (2016). Revised hydrogeologic framework of the Floridan aquifer system in Florida and parts of Georgia, Alabama, and South Carolina. U.S. Geological Survey Professional Paper 1807, 140 pp. Accessed March 31, 2021 from https://pubs.usgs.gov/pp/1807/pdf/pp1807.pdf
Lower Coastal Plain	Floridan-Aquifer System	Unconsolidated deposits overlying carbonate rocks *surficial deposits are up to ~10 m thick see Fig. 20 by Williams and Kuniansky (2016).	"A thick sequence of carbonate rocks (limestone and dolomite) of Tertiary age comprise the Floridan aquifer system." Text quoted from Miller (1990). "The hydrogeologic units of the NC and SC Coastal Plain have been described in the following publications: Aucott (1996), Winner and Coble (1996), Lautier (1998, 2002, 2006), and Harrelson and Fine (2006). The aquifers consist of layers of permeable sand or carbonate rocks separated by confining units	Miller, J.A. (1990). Ground Water Atlas of the United States: Segment 6, Alabama, Florida, Georgia, South Carolina. U.S. Geological Survey Hydrologic Atlas 730-G, 30 pp. Accessed April 5, 2021 from https://www.nrc.gov/docs/ML1706/ML1706B027.pdf Campbell, B.G., and Coes, A.L., eds. (2010). Groundwater availability in the Atlantic Coastal Plain of North and South Carolina. U.S. Geological Survey Professional Paper 1773, 241 p., 7 pls. https://pubs.usgs.gov/pp/1773/pdf/pp1773.pdf

Aquifer system title	Broader aquifer system title	Classification*	Basis for classification (cited figures are geologic cross-sections in most cases—text often quoted directly from reference in adjacent column (i.e., the column to the right))	Reference(s)
			of silt, clay, or low-permeability carbonate rocks." Text quoted from Campbell and Coes (2010) "The Coastal Plain Province is underlain by a wedge of unconsolidated to poorly consolidated sand, clay, silt, and limestone of Late Cretaceous and younger ages deposited on pre-Cretaceous metamorphic, igneous, and sedimentary rocks. These pre-Cretaceous rocks consist of sedimentary rocks of Triassic age and metamorphic and igneous rocks similar to those found at or near the land surface in the Piedmont (Overstreet and Bell, 1965; Marine and Siple, 1974; Cohn and others, 1977; Daniele and Zietz, 1978). The re-Cretaceous rocks in South Carolina are much less permeable than the overlying sediments composing the aquifers of the Coastal Plain and are separated from them by a weathered mantle of low permeability saprolite (Wait and Davis, 1986)." Text quoted from Aucott (1996).	Aucott, W. R. (1996). Hydrology of the southeastern Coastal Plain aquifer system in South Carolina and parts of Georgia and North Carolina (No. 1410-E). US Geological Survey. https://pubs.usgs.gov/pp/1410e/report.pdf
Ocala Uplift	Floridan Aquifer System	Carbonate rocks interbedded with clastic sedimentary rocks "surficial deposits are thin see Fig. 20 by Williams and Kuniansky (2016)	"A thick sequence of carbonate rocks (limestone and dolomite) of Tertiary age comprise the Floridan aquifer system." Text quoted from Miller (1990). "The top of the Floridan aquifer system has been subjected to extensive karstification across the broad, northwest-trending Ocala uplift or Ocala "platform" in north-central Florida and along the western coast of the peninsula near Dixie and Levy Counties, and extending into parts of west central Florida (fig. 10, pl. 4). Across the Ocala platform, the upper confining unit overlying the limestone of the aquifer has been partly or completely removed resulting in greater karst development at the top of the aquifer system." Text quoted from Williams and Kuniansky (2016), page 44)	Miller, J.A. (1990). Ground Water Atlas of the United States: Segment 6, Alabama, Florida, Georgia, South Carolina. U.S. Geological Survey Hydrologic Atlas 730-G, 30 pp. Accessed April 5, 2021 from https://www.nrc.gov/docs/ML1706/ML17060B027.pdf Williams, L.J., Kuniansky, E.L. (2016). Revised hydrogeologic framework of the Floridan aquifer system in Florida and parts of Georgia, Alabama, and South Carolina. U.S. Geological Survey Professional Paper 1807, 140 pp. Accessed March 31, 2021 from https://pubs.usgs.gov/pp/1807/pdf/pp1807.pdf
Sea Island	Floridan Aquifer System	Unconsolidated deposits overlying carbonate rocks "surficial deposits are up to ~60 m thick see Fig. 20 by Williams and Kuniansky (2016).	"A thick sequence of carbonate rocks (limestone and dolomite) of Tertiary age comprise the Floridan aquifer system." Text quoted from Miller (1990). "Coastal Plain strata in the study area consist of unconsolidated to semiconsolidated layers of sand and clay, and semiconsolidated to very dense, layers of limestone and dolomite. These sedimentary rocks unconformably overlie igneous intrusive rocks and low-grade metamorphic rocks of Paleozoic age, and sedimentary strata and volcanics of Triassic to Early Jurassic age	Miller, J.A. (1990). Ground Water Atlas of the United States: Segment 6, Alabama, Florida, Georgia, South Carolina. U.S. Geological Survey Hydrologic Atlas 730-G, 30 pp. Accessed April 5, 2021 from https://www.nrc.gov/docs/ML1706/ML17060B027.pdf Clarke, J.S., Hacke, C.M., and Peck, M.F. (1990). Geology and ground-water resources of the coastal area of Georgia. Georgia Geologic Survey Bulletin 113, 106 p. Accessed January 25, 2022 via https://epd.georgia.gov/sites/epd.georgia.gov/files/related_files/sit

Aquifer system title	Broader aquifer system title	Classification*	Basis for classification (cited figures are geologic cross-sections in most cases—text often quoted directly from reference in adjacent column (i.e., the column to the right))	Reference(s)
			(Chowans and Williams, 1983).” Text quoted from Clark et al. (1990), page 8. Upper and Lower Brunswick aquifers consist of, “poorly sorted, fine to coarse, slightly phosphatic and dolomitic” sand. Text quoted from Clark et al. (1990), page 26.	e_page/B-113.pdf
Tifton Upland	Floridan Aquifer System	Carbonate rocks interbedded with clastic sedimentary rocks. “surficial deposits are thin see Fig. 20 by Williams and Kuniansky (2016); thick sedimentary sequences (i.e., mudstone) overlie the carbonate sequences in much of this region	“A thick sequence of carbonate rocks (limestone and dolomite) of Tertiary age comprise the Floridan aquifer system.” Text quoted from Miller (1990). “In the Tifton Upland, the Ocala Limestone is deeply buried by thick Miocene and Oligocene deposits” Text quoted from Kellam and Gordey, (1990), page 9. Page 14 of Owen (1963) describes stratigraphy of this area with 50 feet of mudstone over Ocala limestone.	Miller, J.A. (1990). Ground Water Atlas of the United States: Segment 6, Alabama, Florida, Georgia, South Carolina. U.S. Geological Survey Hydrologic Atlas 730-G, 30 pp. Accessed April 5, 2021 from https://www.nrc.gov/docs/ML1706/ML17060B027.pdf Kellam, M. F., & Gorday, L. L. (1990). Hydrogeology of the Gulf Trough–Apalachicola Embayment area. Georgia: Georgia Geologic Survey Bulletin, 94, 74 pp. https://epd.georgia.gov/sites/epd.georgia.gov/files/related_files/exit_e_page/B-94.pdf Owen, V. (1963). Geology and Ground Water Resources of Mitchell County, Georgia. Georgia Geologic Survey Information Circular 24, 40 pp. Accessed from: https://epd.georgia.gov/document/publication/ic-24-geology-and-ground-water-resources-mitchell-county-georgia-1963/download
Vidalia Upland	Floridan Aquifer System	Unconsolidated deposits overlying carbonate rocks. “surficial deposits are up to ~30 m thick see Fig. 20 by Williams and Kuniansky (2016).	“A thick sequence of carbonate rocks (limestone and dolomite) of Tertiary age comprise the Floridan aquifer system.” Text quoted from Miller (1990).	Miller, J.A. (1990). Ground Water Atlas of the United States: Segment 6, Alabama, Florida, Georgia, South Carolina. U.S. Geological Survey Hydrologic Atlas 730-G, 30 pp. Accessed April 5, 2021 from https://www.nrc.gov/docs/ML1706/ML17060B027.pdf
Catahoula Area	Gulf Coast Regional Aquifer System	Partly semi-consolidated dipping layered clastic sedimentary sequences	“Geologic units at land surface in the study area are sediments of Miocene age consisting of a complex series of alternating and lenticular beds of sand and clay, and other sediments of Pliocene and younger age. The beds of Miocene age dip to the southwest 30 to 100 ft/mi.” Text quoted directly from Halford and Barber (1995).	Halford, K. J., Barber, N. L. (1995). Analysis of ground water flow in the Catahoula aquifer system in the vicinity of Laurel and Hattiesburg, Mississippi. U.S. Geological Survey Water Resources Investigations Report 94-4219, 78 pp. Accessed March 31, 2021 from https://pubs.usgs.gov/wri/1994/4219/report.pdf
Houston-Galveston Area	Gulf Coast Regional Aquifer System	Partly semi-consolidated dipping layered clastic sedimentary	Fig. 2 by Braun et al. (2019); “The three primary aquifers in the Gulf Coast aquifer system in the Houston-Galveston region study area (the Chicot, Evangeline, and Jasper aquifers) are composed of laterally discontinuous deposits of gravel, sand, silt, and clay	Braun, C.L., Ramage, J.K., Shah, S.D. (2019). Status of groundwater level altitudes and long-term groundwater level changes in the Chicot, Evangeline, and Jasper aquifers, Houston-Galveston region, Texas (2019). U.S. Geological Survey Scientific

Aquifer system title	Broader aquifer system title	Classification*	Basis for classification (cited figures are geologic cross-sections in most cases — text often quoted directly from reference in adjacent column (i.e., the column to the right))	Reference(s)
		sequences	(Baker, 1979). The percentage of clay and other fine-grained, clastic material generally increases downdip (Baker, 1979). The uppermost aquifer, the Chicot aquifer, is contained in Holocene and Pleistocene-age (Quaternary-age) sediments; the underlying Evangeline aquifer is contained in Pliocene and Miocene-age sediments; and the most deeply buried of the three aquifers, the Jasper aquifer, is contained in Miocene-age sediments (fig. 2) (Baker, 1979, 1986). Hydrogeologic units in the study area include the Chicot aquifer, Evangeline aquifer, Burkeville confining unit, Jasper aquifer, and Catahoula confining system" and "Through time, geologic and hydrologic processes created accretionary sediment wedges (stacked sequences of sediments) more than 7,600 ft thick at the coast..." Text quoted directly from Braun et al. (2019).	Investigations Report 2019–5089, 18 pp. https://pubs.usgs.gov/sir/2019/5089/sir20195089.pdf
Lafayette Area	Gulf Coast Regional Aquifer System	Poorly semi-consolidated dipping layered-elastic sedimentary sequences to confining unit ~200 ft in thickness is shown to be at land surface in Fig. 3 of Louisiana Department of Transportation and Development (2011)	Fig. 3 of Louisiana Department of Transportation and Development (2011); "The upper sand of the Chicot aquifer system in Lafayette Parish consists mostly of coarse sand, grading to gravel near the base of individual beds. The sand beds are generally several hundred feet thick and are separated in places by thick, discontinuous clays." Text quoted directly from Louisiana Department of Transportation and Development (2011).	Louisiana Department of Transportation and Development (2011). Water Resources of Lafayette Parish. U.S. Geological Survey Fact Sheet 2010–3048, 6 pp. Accessed May 27, 2021 from https://pubs.usgs.gov/fs/2010/3048/pdf/FS2010-3048.pdf
Southern Hills	Gulf Coast Regional Aquifer System	Poorly semi-consolidated dipping layered-elastic sedimentary sequences	Fig. 2 by Boswell (1979); "The Southern Hills aquifer system as named in this report is comprised of a gulfward thickening and dipping, complexly interbedded, series of sandy and clayey formations that generally range in age from Pleistocene or Pliocene at the top, to Miocene at the base. (See figs. 3, 4, 5, and 6; table 1.) The base of the aquifer system generally has been considered to be the base of freshwater diagrammatically shown in fig. 6) within the sedimentary sequence, defined as 250 mg/L or less chloride concentration in southeastern Louisiana and 1,000 mg/L or less dissolved solids concentration in southern Mississippi. Based on these chemical concentrations, the aquifer system attains a thickness of more than 2,500 ft in the southernmost part of the study area..." Text quoted from Buono (1983).	Boswell, E.H. (1979). The Citronelle aquifers in Mississippi. U.S. Geological Survey Water Resources Investigations Report 78-131, 4 plate. Accessed March 31, 2021 from https://pubs.usgs.gov/wri/1978/0131/plate-1.pdf Buono, A. (1983). The Southern Hills Regional Aquifer System of Southeastern Louisiana And Southwestern Mississippi. U.S. Geological Survey Water Resources Investigations Report 83-4189, 44 pp. Accessed March 31, 2021 from https://pubs.usgs.gov/wri/1983/4189/report.pdf
Central High	High Plains	Moderately thick	Fig. 7 of Gutentag (1994); "The High Plains aquifer consists	Gutentag, E. D., Heimes, F. J., Krothe, N. C., Luekey, R. R.,

Aquifer system title	Broader aquifer system title	Classification*	Basis for classification (cited figures are geologic cross-sections in most cases—text often quoted directly from reference in adjacent column (i.e., the column to the right))	Reference(s)
Plains		unconsolidated aquifer overlying layered-clastic sedimentary sequences	mainly of one or more hydraulically connected geologic units of late-Tertiary or Quaternary age. The principal geologic units in the High Plains aquifer are shown in figure 3 and summarized in the generalized geologic section presented in table 1. The upper Tertiary rocks consist of the Brule Formation (of the White River Group), Arikaree Group, and Ogallala Formation. The Quaternary deposits consist of alluvial, dune sand, and valley-fill deposits." Text quoted from Gutentag et al. (1984); see also their Table 1 with thicknesses for each geologic unit.	Weeks, J. B. (1984). Geohydrology of the High Plains aquifer in parts of Colorado, Kansas, Nebraska, New Mexico, Oklahoma, South Dakota, Texas, and Wyoming (No. 1400-B). Accessed February 10, 2021 from https://pubs.usgs.gov/pp/1400b/report.pdf
Northern High Plains	High Plains	Moderately thick unconsolidated aquifer overlying layered-clastic sedimentary sequences	Fig. 50 of Miller and Appel (1997): "The Ogallala Formation is generally coarser and more permeable than the older underlying units and extends over most of the study area (fig. 6, table 2). The Ogallala Formation is a heterogeneous deposit of interlayered stream sediments, lakebeds, and eolian sand, silt, and clay. The Ogallala Formation varies greatly in particle size and physical character over short distances (Cannia and others, 2006). The maximum thickness of the Ogallala Formation is about 984 ft (Hobza and others, 2012; Flynn and Stanton, 2018). Sediments of the Ogallala Formation form the thickest part of the Northern High Plains aquifer. Sediments of the Ogallala Formation are less coarse than the overlying Quaternary alluvial and valley-fill deposits; gravel is not abundant within the Ogallala Formation (Lawton, 1984). Unconsolidated Quaternary-age sedimentary deposits overlie the older aquifer units" Text quoted directly from Peterson et al. (2020).	Miller, J. A., Appel, C. L. (1997). Ground Water Atlas of the United States: Kansas, Missouri, and Nebraska HA. United States Geological Survey Report 730-D. Accessed February 10, 2021 from Peterson, S. M., Traylor, J. P., Guira, M. (2020). Groundwater Availability of the Northern High Plains Aquifer in Colorado, Kansas, Nebraska, South Dakota, and Wyoming. U.S. Geological Survey Professional Paper 1864, 57 p., Accessed November 28, 2021 from https://pubs.usgs.gov/pp/1864/pp1864.pdf
Southern High Plains	High Plains	Moderately thick unconsolidated aquifer overlying layered-clastic sedimentary sequences	Fig. 15 of Blandford (2008): "The study area is underlain mainly by the Tertiary Ogallala Formation and the Quaternary Blackwater Draw and Tule Formations. The Ogallala Formation ranges in thickness from 0 to more than 500 feet and consists of fluvial gravel, sand, and silt, and eolian sand and silt." Text quoted directly from Blandford et al. (2008).	Blandford, T. N., Kuchanur, M., Standen, A., Ruggiero, R., Calhoun, K. C., Kirby, P., Shah, G. (2008). Groundwater Availability Model of the Edwards-Trinity (High Plains) Aquifer in Texas and New Mexico. Texas Water Development Board Report, 282 pp.
Albuquerque Basin	Middle Rio Grande	Thick alluvium and poorly/semi-consolidated sedimentary formations	Fig. 3 of Hawley (1995): "The Santa Fe Group aquifer system is divided into three parts: the upper (from less than 1,000 to 1,500 feet thick), middle (from 250 to 9,000 feet thick), and lower (from less than 1,000 to 3,500 feet thick). In places, the upper part and/or the middle part of the aquifer has eroded away. Much of the lower part may have low permeability and poor water chemistry; thus, ground water is mostly withdrawn from the upper and middle parts of the aquifer. Only about the upper 2,000 feet of	Hawley, J. W., Haase, C. S., Lozinsky, R. P. (1995). An Underground View of the Albuquerque Basin. Report No. CONF-9414293-TRN-IM9704%261-37-55. Accessed November 28, 2021 from https://nmwrri.nmsu.edu/wp-content/uploads/2015/watcon/proc39/Hawley.pdf Bartolino, J. R., Cole, J. C. (2002). Ground-water resources of the Middle Rio Grande Basin. U.S. Geological Survey Water-

Aquifer system title	Broader aquifer system title	Classification*	Basis for classification (cited figures are geologic cross-sections in most cases—text often quoted directly from reference in adjacent column (i.e., the column to the right))	Reference(s)
			the aquifer is typically used for ground-water withdrawal. Ground water from the Santa Fe Group aquifer system is currently the sole source of water for municipal supply, domestic, commercial, and industrial use in the Middle Rio Grande Basin." Text quoted directly from Bartolino and Cole (2002).	Resources Circular 1222, 145 pp. Accessed November 28, 2021 from https://pubs.usgs.gov/circ/2002/circ1222/pdf/circ1222.pdf
Española Basin	Middle Rio Grande	Thick alluvium and poorly/semi-consolidated sedimentary formations	Fig. 2 of Grauch et al. (2009): "The southern Española Basin consists of a westward and northward thickening wedge of rift fill, composed primarily of Santa Fe Group sediments, that serves as the principal aquifer for the city of Santa Fe and surrounding areas." Text quoted from Grauch et al. (2009). "The basin fill consists mainly of Quaternary alluvium along washes and river channels underlain by Miocene to Pliocene sediments and sedimentary rocks that belong to the Santa Fe Group. The thickness of the Santa Fe Group ranges from 0 m at the contact with the Sangre de Cristo Mountains to as much as 2,000–3,000 m in the central and western parts of the basin. The Miocene Tesuque Formation is the dominant member of the Santa Fe Group in the Española Basin. It is the principal aquifer throughout most of the basin, and most wells in the basin fill sampled in this study are completed within it. The formation consists of variably consolidated sediments with clast sizes ranging from clay/claystone to sand/sandstone and gravel/pebble conglomerate." Text quoted from Manning (2011). "Primary aquifers in the Española Basin are contained within the Tertiary-Quaternary Santa Fe Group. Basin-fill aquifers of the Santa Fe Group are the principal groundwater resource for the cities of Santa Fe, Española, and six Pueblo nations. The Santa Fe Group thickens to the west and north, ranging from ~250 feet thick south of Santa Fe to greater than 10,000 feet beneath the Pajarito Plateau..." Text quoted from Land (2016).	Grauch, V.J.S., Phillips, J.D., Koning, D.J., Johnson, P.S., Bankey, V. (2009). Geophysical interpretations of the southern Española Basin, New Mexico, that contribute to understanding its hydrogeologic framework: U.S. Geological Survey Professional Paper 1761, 88 p. Accessed November 28, 2021 from https://pubs.usgs.gov/pp/1761/download/pdf/PP1761.pdf Manning, A. H. (2011). Mountain block recharge, present and past, in the eastern Española Basin, New Mexico, USA. Hydrogeology Journal, 19, 379–397. Land, L. (2016). Overview of Fresh and Brackish Water Quality in New Mexico. Open file Report 583. 4 pp. Accessed February 17, 2021 from https://geoinfo.nmt.edu/resources/water/amp/brochures/BWA/Espanola_Basin_FBWQNM.pdf
San Luis Valley	Middle Rio Grande	Thick alluvium and poorly/semi-consolidated sedimentary formations	Fig. 3 of Bexfield & Anderholm (2014): "The basin fill deposits of the Santa Fe and Los Pinos Formations, which range in age from Oligocene to Pliocene, are as thick as 10,000 ft in the eastern subbasin of the Alamosa Basin (McCalpin, 1996). The Los Pinos Formation forms an eastward thickening wedge along the eastern border of the San Juan Mountains, consisting of sandy gravel with interbedded volcanoclastic sandstone and tuffaceous material. The Santa Fe Formation, which is predominant in the eastern part of the Alamosa Basin and intertongues with the Los Pinos Formation, consists of buff to pinkish-orange clays with	Bexfield, L. and Anderholm, S. (2014). Conceptual Understanding and Groundwater Quality of Selected Basin-Fill Aquifers in the Southwestern United States: Section 10. Conceptual Understanding and Groundwater Quality of the Basin-Fill Aquifer in the San Luis Valley, Colorado and New Mexico. Geological Survey Professional Paper 1781, 26 pp. Accessed April 22, 2021 from https://pubs.usgs.gov/pp/1781/pdf/pp1781_section10.pdf

Aquifer system title	Broader aquifer system title	Classification*	Basis for classification (cited figures are geologic cross-sections in most cases—text often quoted directly from reference in adjacent column (i.e., the column to the right))	Reference(s)
			interbedded, poorly to moderately sorted silty sands" and "In general terms, the groundwater system of the Alamosa Basin includes two main aquifers—the shallow, unconfined aquifer that is present across the entire basin, and the deeper, confined aquifer that is present everywhere except along the basin margins. Although the division between the two aquifers usually is defined by the top of the uppermost blue clay or fine-grained sand in the Alamosa Formation, Hearne and Dewey (1988) emphasize that groundwater conditions in the basin are complex because the overall aquifer system is really a heterogeneous mixture of aquifers and leaky confining beds, each of limited areal extent." Text quoted from Bexfield and Anderholm (2014).	
Central Mississippi Embayment	Mississippi Embayment	Moderately thick unconsolidated aquifer overlying layered clastic sedimentary sequences	Figs. 27, 68 and 69 of Renken (1998): "The Mississippi embayment aquifer system extends eastward from Arkansas to northwestern Mississippi and comprises six aquifers that crop out as an arcuate band of poorly consolidated to unconsolidated, bedded sand, silt and clay (fig. 66). Geologic units of the aquifer system range from Late Cretaceous to middle Eocene in age. In southern Mississippi and central Louisiana, an extensive, thick, clay confining unit, the Vicksburg-Jackson confining unit, separates the Mississippi embayment aquifer system from the overlying Oligocene and younger water-yielding strata of the coastal lowlands aquifer system. In the embayed part of the Gulf Coastal Plain of eastern Arkansas, northeastern Louisiana, and northwestern Mississippi, the southward-dipping strata of the Mississippi embayment aquifer system are hydraulically connected to the Mississippi River Valley alluvial aquifer. The geologic formations and groups that compose the Mississippi embayment aquifer system thicken greatly in southern Mississippi and Louisiana (fig. 67) where large volumes of sediment were deposited by streams that emptied into the ancestral Gulf of Mexico. The Mississippi embayment aquifer system ranges in thickness from a featheredge to more than 6,000 feet (fig. 67). The aquifer system thickens eastward and westward from its updip limits toward the axis of the Mississippi Embayment. The aquifer system is thickest in south-central Louisiana and southwestern Mississippi. Three of the system's six aquifers, the upper and the middle Claiborne and the lower Claiborne-upper Wilcox aquifers, become increasingly clayey and pinch out to the south (fig. 68). Some of the clayey confining units pinch out northward as they become increasingly sandy and more permeable." Text quoted directly from Renken (1998).	Renken, R. A. (1998). Groundwater Atlas of the United States: Arkansas, Louisiana, Mississippi (HA-730-F). U.S. Geological Survey Hydrologic Investigations Atlas 730-F, 30 pp. Accessed November 28, 2021 from https://pubs.usgs.gov/ha/730f/report.pdf

Aquifer system title	Broader aquifer system title	Classification*	Basis for classification (cited figures are geologic cross-sections in most cases—text often quoted directly from reference in adjacent column (i.e., the column to the right))	Reference(s)
Eastern Mississippi Embayment	Mississippi Embayment	Poorly/semi-consolidated dipping layered-clastic sedimentary sequences	Figs. 68 and 69 of Renken (1998): "The Mississippi-embayment aquifer system extends eastward from Arkansas to northwestern Mississippi and comprises six aquifers that crop out as an arcuate band of poorly consolidated to unconsolidated, bedded sand, silt and clay (fig. 66). Geologic units of the aquifer system range from Late Cretaceous to middle Eocene in age. In southern Mississippi and central Louisiana, an extensive, thick, clay confining unit, the Vicksburg-Jackson confining unit, separates the Mississippi embayment aquifer system from the overlying Oligocene and younger water-yielding strata of the coastal lowlands aquifer system. In the embayed part of the Gulf Coastal Plain of eastern Arkansas, northeastern Louisiana, and northwestern Mississippi, the southward-dipping strata of the Mississippi embayment aquifer system are hydraulically connected to the Mississippi River Valley alluvial aquifer. The geologic formations and groups that compose the Mississippi embayment aquifer system thicken greatly in southern Mississippi and Louisiana (fig. 67) where large volumes of sediment were deposited by streams that emptied into the ancestral Gulf of Mexico. The Mississippi embayment aquifer system ranges in thickness from a featheredge to more than 6,000 feet (fig. 67). The aquifer system thickens eastward and westward from its updip limits toward the axis of the Mississippi Embayment. The aquifer system is thickest in south-central Louisiana and southwestern Mississippi. Three of the system's six aquifers, the upper and the middle Claiborne and the lower Claiborne-upper Wilcox aquifers, become increasingly clayey and pinch out to the south (fig. 68). Some of the clayey confining units pinch out northward as they become increasingly sandy and more permeable." Text quoted directly from Renken (1998).	Renken, R. A. (1998). Groundwater Atlas of the United States: Arkansas, Louisiana, Mississippi (HA-730-F). U.S. Geological Survey Hydrologic Investigations Atlas 730-F, 30 pp. Accessed November 28, 2021 from https://pubs.usgs.gov/ha/730f/report.pdf
Western Mississippi Embayment	Mississippi Embayment	Poorly/semi-consolidated dipping layered-clastic sedimentary sequences	Figs. 68 and 69 of Renken (1998): "The Mississippi-embayment aquifer system extends eastward from Arkansas to northwestern Mississippi and comprises six aquifers that crop out as an arcuate band of poorly consolidated to unconsolidated, bedded sand, silt and clay (fig. 66). Geologic units of the aquifer system range from Late Cretaceous to middle Eocene in age. In southern Mississippi and central Louisiana, an extensive, thick, clay confining unit, the Vicksburg-Jackson confining unit, separates the Mississippi embayment aquifer system from the overlying Oligocene and younger water-yielding strata of the coastal lowlands aquifer system. In the embayed part of the Gulf Coastal Plain of eastern Arkansas, northeastern Louisiana, and northwestern Mississippi, the southward-dipping strata of the Mississippi embayment aquifer system are hydraulically connected to the Mississippi River Valley alluvial aquifer. The geologic formations and groups that compose	Renken, R. A. (1998). Groundwater Atlas of the United States: Arkansas, Louisiana, Mississippi (HA-730-F). U.S. Geological Survey Hydrologic Investigations Atlas 730-F, 30 pp. Accessed November 28, 2021 from https://pubs.usgs.gov/ha/730f/report.pdf

Aquifer system title	Broader aquifer system title	Classification*	Basis for classification (cited figures are geologic cross-sections in most cases—text often quoted directly from reference in adjacent column (i.e., the column to the right))	Reference(s)
			the Mississippi embayment aquifer system thicken greatly in southern Mississippi and Louisiana (fig. 67) where large volumes of sediment were deposited by streams that emptied into the ancestral Gulf of Mexico. The Mississippi embayment aquifer system ranges in thickness from a featheredge to more than 6,000 feet (fig. 67). The aquifer system thickens eastward and westward from its updip limits toward the axis of the Mississippi Embayment. The aquifer system is thickest in south-central Louisiana and southwestern Mississippi. Three of the system's six aquifers, the upper and the middle Claiborne and the lower Claiborne upper Wilcox aquifers, become increasingly clayey and pinch out to the south (fig. 68). Some of the clayey confining units pinch out northward as they become increasingly sandy and more permeable." Text quoted directly from Renken (1998).	
Delmarva Peninsula	North-Atlantic Coastal Plain	Poorly/semi-consolidated dipping layered clastic sedimentary sequences	Fig. 21 of Sanford (2012): "The Delmarva Peninsula is underlain by Coastal Plain sediments that are Cretaceous through Quaternary in age and alternate between fluvial and marine sands, silts, and clays" text quoted directly from Sanford et al. (2012).	Sanford, W. E., Pope, J. P., Selnick, D. L., Stumvoll, R. F., (2012). Simulation of groundwater flow in the shallow aquifer system of the Delmarva Peninsula, Maryland and Delaware. US Geological Survey Open File Report 2012-1140, 68 pp. Accessed November 20, 2021 from https://pubs.usgs.gov/of/2012/1140/pdf/OFR_2012-1140.pdf
Maryland Western Shores	North-Atlantic Coastal Plain	Poorly/semi-consolidated dipping layered clastic sedimentary sequences	Figs. 4-6 of Vroblesky (1991): "The Coastal Plain of Maryland, Delaware, and the District of Columbia encompasses an area of about 8,500 mi² (fig. 4). The sediments extend approximately another 75 mi eastward from the present coastline to the edge of the Continental Shelf. The Coastal Plain is underlain by a series of eastward- and southeastward-dipping deposits of mostly unconsolidated gravel, sand, silt, and clay." Text quoted directly from Vroblesky et al. (1991).	Vroblesky, D.A., Fleck, W.B. (1991). Hydrogeologic Framework of the Coastal Plain of Maryland, Delaware, and the District of Columbia. U.S. Geological Survey Professional Paper 1404-E, 52 pp. Accessed April 1, 2021 from https://pubs.usgs.gov/pp/1404e/report.pdf
New Jersey Coastal Plain	North-Atlantic Coastal Plain	Poorly/semi-consolidated dipping layered clastic sedimentary sequences	Fig. 4 of Gordon (2021): "...consists of a southeastward dipping and thickening wedge of unconsolidated deposits of sand, silt, and clay of Cretaceous to Neogene age underlain by basement rocks and overlain by a veneer of locally occurring Quaternary sediments (fig. 1, table 1). Coastal Plain sediments were deposited in various shelf, marginal marine, near shore or coastal beach, and deltaic environments, the extent of which fluctuated in response to relative changes in sea level. Units composed of distinctly less permeable sediments (predominantly clays and fine-grained silts) form the confining units, and coarser, more permeable sand and gravel units, which readily produce water, form the aquifers (fig. 1). These deposits are less than 50 ft thick along the western limit of the Coastal Plain (Fall Line) and thicken	Gordon, A.D., Carleton, G.B., Rosman, R. (2021). Water-level conditions in the confined aquifers of the New Jersey Coastal Plain, 2013. U.S. Geological Survey Scientific Investigations Report 2019-5146, 104 p., 0 pl., Accessed April 1, 2021 from https://pubs.usgs.gov/sir/2019/5146/sir20195146.pdf

Aquifer system title	Broader aquifer system title	Classification*	Basis for classification (cited figures are geologic cross-sections in most cases—text often quoted directly from reference in adjacent column (i.e., the column to the right))	Reference(s)
			to more than 6,500 ft in southern Cape May County." Text quoted directly from Gordon et al. (2024).	
North Carolina and Virginia Coastal Plain	North Atlantic Coastal Plain	Highly semi-consolidated dipping layered elastic sedimentary sequences	Plates 2–16 of Winner Jr. and Coble (1989): "The study area of the Northern Atlantic Coastal Plain Regional Aquifer System, which is one of the studies of the U.S. Geological Survey Regional Aquifer System Analysis (RASA) program borders the east coast of the United States between Long Island, New York, and the North Carolina–South Carolina State line. This regional aquifer system is an eastward-dipping and eastward-thickening wedge of sedimentary rocks ranging in age from Early Cretaceous to Holocene that were deposited mostly on metamorphic and igneous crystalline basement rocks; locally, the sedimentary wedge lies on low permeability red beds of early Mesozoic age. The wedge is a featheredge at the western limit of the Coastal Plain and is more than 10,000 feet thick at Cape Hatteras." Text quoted directly from Winner Jr. and Coble (1989).	Winner Jr., M.D., Coble, R.W. (1989). Hydrogeologic framework of the North Carolina Coastal Plain aquifer system. U.S. Geological Survey Report 87-690, 167 pp. Accessed April 1, 2024 from https://pubs.usgs.gov/of/4987/0690/report.pdf
Powder River Basin	Northern Great Plains	Consolidated elastic sedimentary rock formations	Figs. 3 and 4 of Long et al. (2018): "The upper Fort Union aquifer is as thick as 1,920 ft and composed of crossbedded light yellow to light yellow-gray sandstone, sandy mudstone, gray shale, carbonaceous shale, and thick coal beds and associated clinker deposits (permeable rocks created by the natural burning of coal beds)," and "Present throughout the central part of the Williston Basin, the middle Fort Union hydrogeologic unit thins toward the northeast and is not present in the northeastern one-third of the basin (fig. 4). Composed of as much as 520 ft of thickness of alternating beds of sandstone, siltstone, mudstone, claystone, and lignite, rocks in the middle Fort Union hydrogeologic unit generally are finer grained and darker colored than the overlying upper Fort Union aquifer and underlying lower Fort Union aquifer," and "The upper Hell Creek hydrogeologic unit is composed of as much as 740 ft of thickness of alternating layers of gray and brown mudstone, siltstone, sandstone, and sparse lignite beds deposited by meandering streams with point bars and channel plugs." Text quoted directly from Long et al. (2018).	Long, A.J., Thamke, J.N., Davis, K.W., Bartos, T.T. (2018). Groundwater availability of the Williston Basin, United States and Canada: U.S. Geological Survey Professional Paper 1841, 54 pp. Accessed November 29, 2021 from https://pubs.usgs.gov/pp/1841/pp1841.pdf
Williston Basin	Northern Great Plains	Glacial drift overlying consolidated elastic sedimentary sequences—glacial drift common in northern portion of	Figs. 3 and 4 of Long et al. (2018): "From shallowest (youngest) to deepest (oldest), the three principal aquifer systems are the glacial, lower Tertiary, and Upper Cretaceous aquifer systems, with the latter two contained within bedrock (fig. 2). These three principal aquifer systems are hydraulically separated from deeper aquifers by a basal confining unit composed primarily of 800–	Long, A.J., Thamke, J.N., Davis, K.W., Bartos, T.T. (2018). Groundwater availability of the Williston Basin, United States and Canada: U.S. Geological Survey Professional Paper 1841, 54 pp. Accessed November 29, 2021 from https://pubs.usgs.gov/pp/1841/pp1841.pdf

Aquifer system title	Broader aquifer system title	Classification*	Basis for classification (cited figures are geologic cross-sections in most cases—text often quoted directly from reference in adjacent column (i.e., the column to the right))	Reference(s)
		area	3,000-ft of low-permeability marine shale of Upper Cretaceous age; the altitude of the top is shown in figure 3. The bowl-shaped structure and layered geology of the Williston Basin results in bedrock hydrogeologic units that are exposed to the land surface or are in contact with the overlying glacial aquifer system near the outer parts of the basin (figs. 1, 4). The glacial aquifer system overlies parts of the lower Tertiary and Upper Cretaceous aquifer systems in the northern part of the Williston Basin (fig. 1)." and "Thickness of the glacial aquifer system in the Williston Basin varies locally; maximum thickness is estimated to be about 750 ft" and "The upper Fort Union aquifer is as thick as 1,920 ft and composed of crossbedded light yellow to light yellow-gray sandstone, sandy mudstone, gray shale, carbonaceous shale, and thick coal beds and associated clinker deposits (permeable rocks created by the natural burning of coal beds)." and "Present throughout the central part of the Williston Basin, the middle Fort Union hydrogeologic unit thins toward the northeast and is not present in the northeastern one-third of the basin (fig. 4). Composed of as much as 520 ft of thickness of alternating beds of sandstone, siltstone, mudstone, claystone, and lignite, rocks in the middle Fort Union hydrogeologic unit generally are finer-grained and darker colored than the overlying upper Fort Union aquifer and underlying lower Fort Union aquifer." and "The upper Hell Creek hydrogeologic unit is composed of as much as 740 ft of thickness of alternating layers of gray and brown mudstone, siltstone, sandstone, and sparse lignite beds deposited by meandering streams with point bars and channel plugs." Text quoted directly from Long et al. (2018).	
Eastern Cambrian-Ordovician Aquifers	Northern Midwest Aquifer System	Glacial drift overlying consolidated elastic sedimentary sequences "glacial drift is thin or absent in part of this eubarea (see "Driftless area" in Fig. 30 by Olcott (1992))	Figs. 12-18 by Olcott (1992); "The surficial aquifer system consists of permeable, sorted and stratified sand and gravel deposits of glacial origin (aquifers) that commonly are interbedded with less permeable till (confining units). Because of its permeability and extent throughout most of the segment, the surficial aquifer system is hydraulically connected with nearly all bedrock aquifers. The bedrock aquifers are exposed at the land surface only in small areas; generally, they are covered by the surficial aquifer system, which provides an important component of storage from which water percolates downward to the underlying aquifers." and "The Cambrian-Ordovician aquifer system consists of three sandstone aquifers separated by partial confining units and it is capped by the Maquoketa confining unit." Text quoted directly from Olcott (1992).	Olcott, P.G. (1992). Groundwater Atlas of the United States: Segment 9 Iowa, Michigan, Minnesota, Wisconsin. U.S. Geological Survey Hydrologic Atlas 730-J, 33 pp. Accessed April 12, 2024 from https://pubs.usgs.gov/ha/730j/report.pdf

Aquifer system title	Broader aquifer system title	Classification*	Basis for classification (cited figures are geologic cross-sections in most cases—text often quoted directly from reference in adjacent column (i.e., the column to the right))	Reference(s)
Eastern Silurian-Devonian Aquifers	Northern Midwest Aquifer System	Glacial drift overlying carbonate rocks *glacial drift is tens of meters in much of this subarea (see Fig. 30 by Olcott (1992))	Figs. 12-18 by Olcott (1992); "The Silurian-Devonian aquifer consists mostly of dolomite and limestone in which fracture permeability has been enhanced by solution and extensive karst development." Text quoted from Olcott (1992).	Olcott, P.G. (1992). Groundwater Atlas of the United States: Segment 9 Iowa, Michigan, Minnesota, Wisconsin. U.S. Geological Survey Hydrologic Atlas 730-J, 33 pp. Accessed April 12, 2021 from https://pubs.usgs.gov/ha/730j/report.pdf
Mississippian-Silurian-Devonian Carbonates	Northern Midwest Aquifer System	Glacial drift overlying carbonate rocks *glacial drift is tens of meters in much of this subarea (see Fig. 30 by Olcott (1992))	Figs. 12-18 by Olcott (1992); "The Mississippian aquifer in Iowa consists of dolomite and forms the bedrock surface in the central and the southeastern parts of the State (figs. 7 and 15). In southwestern Iowa, the aquifer is overlain and underlain by shale confining units (fig. 8)" and "The Silurian-Devonian aquifer consists mostly of dolomite and limestone in which fracture permeability has been enhanced by solution and extensive karst development." Text quoted from Olcott (1992).	Olcott, P.G. (1992). Groundwater Atlas of the United States: Segment 9 Iowa, Michigan, Minnesota, Wisconsin. U.S. Geological Survey Hydrologic Atlas 730-J, 33 pp. Accessed April 12, 2021 from https://pubs.usgs.gov/ha/730j/report.pdf
Northeast Missouri Carbonates	Northern Midwest Aquifer System	Glacial drift overlying carbonate rocks	"Limestone and sandstone in rocks of Mississippian age." Text quoted from Miller (1995). "The uppermost aquifer is in Mississippian carbonate rocks; stratigraphically equivalent carbonate rocks in northern Missouri are called the Mississippian aquifer..." Text quoted from Miller and Appel (1997). "Glacial drift overlies much of the bedrock throughout the region." text (pertaining to their defined "Area 2" is quoted from the following Missouri Department of Natural Resources Division of Geology and Land Survey webpage: http://agebb.missouri.edu/drought/water/MissouriGroundwaterProvincesandAquiferCharacteristics.pdf	Miller, J.A. (1995). Ground Water Atlas of the United States: Segment 10, Illinois, Indiana, Kentucky, Ohio, Tennessee. U.S. Geological Survey Hydrologic Atlas 730-K, 30 pp. Accessed April 11, 2021 from https://pubs.usgs.gov/ha/730k/report.pdf Miller, J.A., Appel, C.L. (1997). Ground Water Atlas of the United States: Segment 3, Kansas, Missouri, Nebraska. U.S. Geological Survey Hydrologic Atlas 730-D, 26 pp. Accessed April 13, 2021 from https://pubs.usgs.gov/ha/730d/report.pdf
Northern Cambrian-Ordovician Aquifers	Northern Midwest Aquifer System	Glacial drift overlying consolidated clastic sedimentary sequences	Figs. 12-18 by Olcott (1992); "The surficial aquifer system consists of permeable, sorted and stratified sand and gravel deposits of glacial origin (aquifers) that commonly are interbedded with less permeable till (confining units). Because of its permeability and extent throughout most of the segment, the surficial aquifer system is hydraulically connected with nearly all bedrock aquifers. The bedrock aquifers are exposed at the land surface only in small areas; generally, they are covered by the surficial aquifer system, which provides an important component of storage from which water percolates downward to the underlying aquifers The Cambrian-Ordovician aquifer system consists of three sandstone aquifers separated by partial confining units and it is	Olcott, P.G. (1992). Groundwater Atlas of the United States: Segment 9 Iowa, Michigan, Minnesota, Wisconsin. U.S. Geological Survey Hydrologic Atlas 730-J, 33 pp. Accessed April 12, 2021 from https://pubs.usgs.gov/ha/730j/report.pdf

Aquifer system title	Broader aquifer system title	Classification*	Basis for classification (cited figures are geologic cross-sections in most cases—text often quoted directly from reference in adjacent column (i.e., the column to the right))	Reference(s)
			capped by the Maquoketa confining unit." Text quoted directly from Olcott (1992).	
Upper Carbonate Aquifer	Northern Midwest Aquifer System	Glacial drift overlying carbonate rocks "glacial drift is thin (<50 ft.) or tens of meters in much of this subarea (see Fig. 30 by Olcott (1992))	Figs. 12–18 by Olcott (1992): "The upper carbonate aquifer, which consists of Ordovician dolomitic shale and dolomite and overlying Devonian dolomite in southeastern Minnesota, is an extremely productive aquifer that extends a short distance into northern Iowa (figs. 7 and 17). The permeability of the aquifer has been substantially enhanced by extensive karst development. The aquifer is underlain by shaly dolomite of Ordovician age that forms the lower part of the Maquoketa confining unit." Text quoted from Olcott (1992).	Olcott, P. G. (1992). Groundwater Atlas of the United States: Segment 9 Iowa, Michigan, Minnesota, Wisconsin. U.S. Geological Survey Hydrologic Atlas 730-J, 33 pp. Accessed April 12, 2021 from https://pubs.usgs.gov/ha/730j/report.pdf
Western Cambrian-Ordovician Aquifers	Northern Midwest Aquifer System	Glacial drift overlying consolidated elastic sedimentary sequences	Figs. 12–18 by Olcott (1992): "The surficial aquifer system consists of permeable, sorted and stratified sand and gravel deposits of glacial origin (aquifers) that commonly are interbedded with less permeable till (confining units). Because of its permeability and extent throughout most of the segment, the surficial aquifer system is hydraulically connected with nearly all bedrock aquifers. The bedrock aquifers are exposed at the land surface only in small areas; generally, they are covered by the surficial aquifer system, which provides an important component of storage from which water percolates downward to the underlying aquifers." and "The Cambrian-Ordovician aquifer system consists of three sandstone aquifers separated by partial confining units and it is capped by the Maquoketa confining unit." Text quoted directly from Olcott (1992).	Olcott, P. G. (1992). Groundwater Atlas of the United States: Segment 9 Iowa, Michigan, Minnesota, Wisconsin. U.S. Geological Survey Hydrologic Atlas 730-J, 33 pp. Accessed April 12, 2021 from https://pubs.usgs.gov/ha/730j/report.pdf
Mesilla Valley	Rincon-Mesilla Valleys	Thick alluvium and poorly/semi-consolidated sedimentary formations	Fig. 4 of Hawley and Lozinsky (1992): "Rift basin fill comprising the Santa Fe Group has a maximum thickness of about 3,000 ft (915 m) in the Mesilla Basin. The lower and middle parts of the Santa Fe form the bulk of the basin fill and were deposited in an internally drained complex of three subbasins that were initially separated by a central (north-trending) intrabasin uplift (Eastern, Northwestern, Southwestern subbasins, and Mid-basin uplift on Plate 1). Eolian sands and fine-grained playa-lake sediments are major lithofacies. These units are locally well-indurated and only produce large amounts of good quality ground water from buried dune sands (lower Santa Fe) in the eastern part of the basin beneath the Mesilla Valley. Poorly consolidated sand and pebble gravel deposits in the middle to upper part of the Santa Fe Group form the most extensive aquifers of the area." Text quoted directly from Hawley and Lozinsky (1992).	Hawley, J. W., Lozinsky, R. P. (1992). Hydrogeologic framework of the Mesilla Basin in New Mexico and western Texas. New Mexico Bureau of Mines and Mineral Resources Open File Report 323, 95 pp. Accessed November 20, 2021 from https://geoinfo.nmt.edu/publications/openfile/downloads/300-399/323/ofr_323.pdf https://geoinfo.nmt.edu/publications/openfile/downloads/300-399/323/ofr_323.pdf

Aquifer system title	Broader aquifer system title	Classification*	Basis for classification (cited figures are geologic cross-sections in most cases—text often quoted directly from reference in adjacent column (i.e., the column to the right))	Reference(s)
West Salt River Basin	Salt Basin	Thick alluvium and poorly/semi-consolidated sedimentary formations	Fig. 2 of Anning (2014): "The basin-fill deposits are divided into upper, middle, and lower units (Brown and Pool, 1989). The lower unit was deposited when the basin was closed and subsiding and it consists of playa, alluvial fan, fluvial, and evaporite deposits. The sediments in the lower unit are generally fine-grained, with coarse-grained facies at the basin margins and at depth, and the unit is further divided into upper and lower parts. The thickness of the lower part exceeds 10,000 ft in the center of the basin, whereas the thickness of the upper part is generally less than 1,000 ft." Text quoted directly from Anning (2014).	Anning, D. (2014). Conceptual Understanding and Groundwater Quality of Selected Basin-Fill Aquifers in the Southwestern United States: Section 7, Conceptual Understanding and Groundwater Quality of the Basin-Fill Aquifer in the West Salt River Valley, Arizona. U.S. Geological Survey Professional Paper 1784, 24 pp. Accessed March 31, 2021 from https://pubs.usgs.gov/pp/1784/pdf/pp1784_section7.pdf
Lower Santa Ynez Valley	Santa Ynez Valley	Thick alluvium and poorly/semi-consolidated sedimentary formations	"Consolidated rocks of Tertiary age underlie the unconsolidated deposits and compose the surrounding hills. These consolidated rocks are marine in origin and consist of relatively impermeable fine-grained deposits of the Foxen Mudstone, Sisquoc Formation, and the Monterey Formation. Wells in these formations yield only small amounts of water from fracture zones and slightly permeable sandstones. Unconsolidated deposits of Pliocene and younger age consist chiefly of sand, gravel, silt, and clay. Wells in unconsolidated deposits yield from several hundred to more than 1,000 gal/min (Miller, 1976). Only unconsolidated deposits are considered in quantitative evaluation of ground-water reserves. Primary ground-water basins in the Santa Ynez River basin are composed of alluvium along the tributaries of the Santa Ynez River, and the main river has several basins of limited volume composed of younger Holocene alluvium." and "The Careaga Sand is a massive body of fine- to coarse-grained marine sand that ranges in thickness from 450 to 1,000 feet. It is overlain by the Paso Robles Formation (Upson and Thomasson, 1951, p. 32). In places, it is locally faulted against the Monterey Formation." Text quoted directly from Hamlin (1985).	Hamlin, S. N. (1985). Ground-water quality in the Santa Rita, Buellton, and Los Olivos hydrologic subareas of the Santa Ynez River basin, Santa Barbara County, California. Water Resources Investigations Report 84-4131, 84 pp. Accessed March 7, 2021 from https://pubs.usgs.gov/wri/1984/4131/report.pdf
Valle de Juarez and Hueco Bolson	Tularosa-Huelco	Thick alluvium and poorly/semi-consolidated sedimentary formations	"The Hueco Bolson water-bearing materials are differentiated into two groups: the clastic-filled graben is composed of fluvial, lacustrine, alluvial fan, and aeolian materials up to 9,000 feet thick; 3 the El Paso-Juarez Valley bolson sediments are overlain by Rio Grande alluvium normally 75 to 100 feet thick, with thicknesses varying locally up to 250 feet." Text quoted directly from Day (1978).	Day, J. C. (1978). International Aquifer Management: The Hueco Bolson on the Rio Grande River. Natural Resources Journal, 163-180.
Boise Valley and Homedale Murphy Area	Western Snake River Plain	Thick alluvium and poorly/semi-consolidated sedimentary	Fig. 9 of Lindholm (1996): "Quaternary-Tertiary sedimentary rocks of the Idaho Group are areally extensive in the western plain. Their composite thickness ranges from zero to as much as 5,000 ft near the Idaho-Oregon border. The Idaho Group is	Lindholm, G. F. (1996). Summary of the Snake River Plain regional aquifer system analysis in Idaho and eastern Oregon. U.S. Geological Survey Professional Paper 1408-A, 59 pp. Accessed March 1, 2021 via

Aquifer system title	Broader aquifer system title	Classification*	Basis for classification (cited figures are geologic cross-sections in most cases—text often quoted directly from reference in adjacent column (i.e., the column to the right))	Reference(s)
		formations	predominantly lacustrine clay and silt, though sand and gravel are common in the Glens Ferry Formation, which is a complex of late Tertiary lacustrine, fluvialite, and flood-plain facies (Malde and Powers, 1962, p. 1206)." Text quoted directly from Lindholm (1996). See also generalized geologic map of Snake River Plain in Fig. 8 of Lindholm (1996)	https://pubs.usgs.gov/pp/1408a/report.pdf
Mountain Home Plateau	Western Snake River Plain	Volcanic (basalt) rocks and alluvial deposits	Fig. 9 of Lindholm (1996); "Ground water occurs in virtually every geologic unit within the study area, as described in table 1. The aquifers primarily consist of basalt, sand and gravel, and crystalline rocks. Water is contained in voids, fractures, joints, and interflow zones in basalt; in intergranular spaces in sand and gravel; and in fractures and weathered zones in crystalline rocks. In the eastern part of the area, the more productive aquifers are basalt flows of the Bruneau Formation of the Idaho Group and in the western part, sand and gravel of the older terrace gravel and the Idaho Group. The Snake River Group also is a source of water in the western part; however, it is above the water table in most places. Water in the aquifers is mostly under water table conditions. Localized perched water zones occur in several places (figs. 5 and 6); the most extensive one is near Mountain Home. Water can also be under artesian conditions where clay, silt, or cemented sand beds constitute confining layers. Water level data are insufficient, however, to discern or map any extensive potentiometric (pressure) surfaces." Text quoted directly from Lindholm (1996). See also generalized geologic map of Snake River Plain in Fig. 8 of Lindholm (1996)	Young, H.W. (1978). Reconnaissance of ground-water resources in the Mountain Home plateau area, southwest Idaho. U.S. Geological Survey Water Resources Investigations Report 77-108, 48 pp. Accessed November 29, 2021 from https://pubs.usgs.gov/wri/1977/0108/report.pdf Lindholm, C. F. (1996). Summary of the Snake River Plain regional aquifer system analysis in Idaho and eastern Oregon. U.S. Geological Survey Professional Paper 1408-A, 59 pp. Accessed March 4, 2021 via https://pubs.usgs.gov/pp/1408a/report.pdf
Antelope Valley	-	Thick alluvium and poorly/semi-consolidated sedimentary formations	Fig. 12 of Stamos (2017); "The west-east section (figs. 6, 12A) shows that the saturated alluvium is thinner to the west and is thickest to the east and that the crest of the ridge of highest basement altitude is between wells 5N9W-18N1 and 5N9W-20P1. Based on the groundwater-level data from September 2014, the saturated thickness of the alluvial deposits near well 5N9W-18N1 was estimated to be about 130 ft." "The westernmost south-north section (figs. 6, 12B) shows that the saturated alluvium is thinnest in this part of the study area and that the maximum depth to basement along the section is near well 6N9W-33G1. The altitude of the basement surface is highest at the north and south ends of the basin where it is exposed, and the groundwater level measured in well 5N9W-33R1 indicates that the water table is probably below or near the basement surface. The driller's log from well 5N9W-33D2 encountered "hard rocks" at about 140 feet below land surface (ft ble)" and "The	Stamos, C. L., Christensen, A. H., Langenheim, V. (2017). Preliminary hydrogeologic assessment near the boundary of the Antelope Valley and El Mirage Valley groundwater basins, California. U.S. Geological Survey Scientific Investigations Report 2017-5065, 56 pp. Accessed March 20, 2021 from https://pubs.usgs.gov/sir/2017/5065/sir20175065.pdf

Aquifer system title	Broader aquifer system title	Classification*	Basis for classification (cited figures are geologic cross-sections in most cases—text often quoted directly from reference in adjacent column (i.e., the column to the right))	Reference(s)
			easternmost south-north section (figs. 6, 12C) shows that the saturated alluvial deposits are more than 2,000 ft thick south of well 5N7W-31J4 but gradually thin to the north to a thickness of about 500 ft; basement outcrops are evidence of the thinning of the alluvial deposits at the north edge of the study area." Text quoted directly from Stamos et al. (2017).	
Big Bear Valley	-	Thick alluvium and poorly/semi-consolidated sedimentary formations	Fig. 6 of Flint and Martin (2012): "Quaternary alluvial deposits overlie the Tertiary sedimentary rocks and basement rocks throughout much of the groundwater basin (fig. 3). For the purposes of this report, the Quaternary alluvial deposits shown on figure 3 were generalized into older alluvium and recent alluvium. The older alluvium consists predominantly of clay and sandy clay with interbedded layers of sand and gravel near Big Bear Lake and coarsens to predominantly sand with some gravel and interbedded layers of silt and clay toward Baldwin Lake." Text quoted directly from Flint and Martin (eds. 2012). See also Fig. 6 (part one one page 12 of Flint and Martin (2012)) where the thickness of alluvium is shown to exceed 500 ft (~150 m) in most areas.	Flint, L.E., Martin, P., eds. (2012). Geohydrology of Big Bear Valley, California: Phase 1—Geologic Framework, Recharge, and Preliminary Assessment of the Source and Age of Groundwater. U.S. Geological Survey Scientific Investigations Report 2012–5400, 112 pp. Accessed November 28, 2021 from https://pubs.usgs.gov/sir/2012/5400/pdf/sir20125400.pdf
Big Chino Valley	-	Thick alluvium and poorly/semi-consolidated sedimentary formations	"Basin fill is composed of up to several thousand feet of unconsolidated, Tertiary to Quaternary sediments interbedded with basalt flows originating from the north, west, and southeast of the valley (Wirt and others, 2005). A thick fine-grained clay unit in the center of the basin (Ostenaa and others, 1993) delineates what once was a playa. This clay unit and other fine-grained units leads to confined or semiconfined aquifer conditions in some places." Text quoted from Kennedy et al. (2019).	Kennedy, J.R., Kahler, L.M., Read, A.L. (2019). Aquifer storage change and storage properties, 2010–2017, in the Big Chino Subbasin, Yavapai County, Arizona. U.S. Geological Survey Scientific Investigations Report 2019–5060, 39 pp. Accessed November 29, 2021 from https://pubs.usgs.gov/sir/2019/5060/sir20195060.pdf
Bighorn Basin	-	Consolidated elastic sedimentary rock formations	"A number of the formations exhibited in the Bighorn basin consist largely of porous rocks, mainly sandstone. These rocks undoubtedly contain water in their underground beds. They outcrop around the outer portion of the basin and extend far up the slopes of the ridges, where in many places they imbibe water from rain and from streams fed by melting snows on the summits of the mountains. In their underground beds they lie immediately beneath impervious shales." Text quoted directly from Fisher (1906).	Fisher, C.A. (1906). Geology and water resources of the Bighorn Basin, Wyoming. US Geological Survey Professional Paper 53, 97 pp. Accessed November 29, 2021 from https://pubs.usgs.gov/pp/0053/report.pdf
Black Hills Uplift	-	Carbonate rocks interbedded with elastic sedimentary rocks	Fig. 15 of Driscoll et al. (2002): "Many of the sedimentary units contain aquifers, both within and beyond the study area. Within the Paleozoic rock interval, aquifers in the Deadwood Formation, Madison Limestone, Minnelusa Formation, and Minnekahta	Driscoll, D.-G., Carter, J. M., Williamson, J. E., Putnam, L. D. (2002). Hydrology of the Black Hills area, South Dakota. US Geological Survey Water Resources Investigations Report 2002-4094, 158 pp. Accessed February 16, 2021 from

Aquifer system title	Broader aquifer system title	Classification*	Basis for classification (cited figures are geologic cross-sections in most cases—text often quoted directly from reference in adjacent column (i.e., the column to the right))	Reference(s)
			Limestone are used extensively. These aquifers are collectively confined by the underlying Precambrian rocks and the overlying Spearfish Formation. Individually, these aquifers are separated by minor confining layers or by low permeability layers within the individual units. In general, ground-water flow in these aquifers is radially outward from the central core of the Black Hills. Although the lateral component of flow generally predominates, the vertical component of flow, and thus leakage between these aquifers, can be extremely variable." Text quoted directly from Driscoll et al. (2002). "The major of the Black Hills area is the Pahasapa Limestone of Mississippian age, the local name applied to an equivalent of the Madison Limestone. These carbonate rocks were deposited during Early Mississippian time, when a great seaway occupied much of the area west of Wyoming. This formation ranges in thickness from 200 m in the northern Hills to less than 100 m in the southern Hills." Text quoted from Back et al. (1983). The compiled groundwater tritium dataset derives from samples of wells located predominantly along the northern, eastern and southern peripheries of the Black Hills, where limestone aquifers are abundant (see Fig. 16 by Driscoll et al. (2002)), thus the classification of the aquifer system as carbonate rocks interbedded with clastic sedimentary rocks.	https://pubs.usgs.gov/wri/wri024094/pdf/wri024094.pdf Back, W., Hanshaw, B. B., Plummer, L. N., Rahn, P. H., Rightmire, C. T., Rubin, M. (1983). Process and rate of dedolomitization: mass transfer and 14C dating in a regional carbonate aquifer. Geological Society of America Bulletin, 94, 1415-1429.
Black Warrior River Aquifer System (Eutaw and McShan Formations and Tuscaloosa Group)	-	Poorty/bentic consolidated dipping layered clastic sedimentary sequences	Fig. 3 of Strom and Mallory (1995): "The upper part of the Eutaw-McShan aquifer has a finer grain size and a larger silt content than the rest of the aquifer and is called the Tombigbee Sand Member. The Tombigbee Sand Member produces little water. The remainder of the Eutaw-McShan aquifer mainly consists of thin beds of fine to medium glauconitic sand (Boeswell, 1963). Analysis of well-log data indicates that total sand thickness within the study area ranges from about 1 foot in the eastern part of the outcrop area to more than 300 feet in the southwestern part and southern part of the study area. The sand is thinnest near the outcrop and generally thickens downdip. An average horizontal hydraulic conductivity value of 12 feet per day, based on the results of 50 aquifer tests, was reported by Slack and Darden (1991)." and "The Eutaw-McShan aquifer and Tuscaloosa aquifer system are part of the larger Southeastern Coastal Plain aquifer system, which is composed primarily of clastic sediments of Cretaceous and Tertiary age." Text quoted directly from Strom and Mallory (1995).	Strom, E. W., Mallory, M. J. (1995). Hydrogeology and simulation of ground-water flow in the Eutaw-McShan Aquifer and in the Tuscaloosa aquifer system in northeastern Mississippi. US Geological Survey Water Resources Investigations Report 94-4223, 89 pp. Accessed November 29, 2021 from https://pubs.usgs.gov/wri/1994/4223/report.pdf

Aquifer system title	Broader aquifer system title	Classification*	Basis for classification (cited figures are geologic cross-sections in most cases—text often quoted directly from reference in adjacent column (i.e., the column to the right))	Reference(s)
Castle Hayne Aquifer	-	Carbonate rocks interbedded with elastic sedimentary rocks	Fig. 1 by Lyke and Coble (1987): "The Castle Hayne aquifer is an eastward-sloping and thickening wedge of limestone and sandstone (fig. 1), which is located in a 12,500 square-mile area in the eastern part of North Carolina." Text quoted directly from Lyke and Coble (1987). Fig. 6 by Eimers et al. (1994): "The observed thickness of the surficial aquifer ranges from 31 to 68 ft (fig. 10)." and "The upper Castle Hayne confining unit overlies the upper Castle Hayne aquifer and is composed of clay and sandy clay at the Air Station. Thin beds of sand are also present." and "The lower Castle Hayne aquifer and confining unit underlie the upper Castle Hayne aquifer and are present everywhere beneath the Air Station. The lower Castle Hayne aquifer is composed of limestone, sandy limestone, calcareous sand, and clay beds. Thin discontinuous stringers of consolidated limestone also are present." Text quoted from Eimers et al. (1994).	Lyke, W.L., Coble, R.W. (1987). Regional study of the Castle Hayne Aquifer of eastern North Carolina. U.S. Geological Survey Open-File Report 87-574, 2 pp. Accessed April 1, 2024 from https://pubs.usgs.gov/of/1987/0574/report.pdf Eimers, J.L., Daniel III, C.C., Coble, R.W. (1994). Hydrogeology and simulation of ground-water flow at U.S. Marine Corps Air Station, Cherry Point, North Carolina, 1987-90. Water Resources Investigations Report 94-4186, 81 pp. Accessed April 1, 2024 from https://pubs.usgs.gov/wri/1994/4186/report.pdf
Coachella Valley	-	Thick alluvium and poorly/semi-consolidated sedimentary formations	"The Coachella Valley is filled with as much as 3,700 m (12,000 ft) of sediments; the upper 610 m (2,000 ft) constitute the aquifer system that is the primary source of groundwater supply (California Department of Water Resources, 1979). The aquifer system consists of a complex unconsolidated to partly consolidated assemblage of gravel, sand, silt, and clay of alluvial and lacustrine origins (fig. 2). Sediments tend to be finer grained (contain more silt and clay) in the southern part of the valley compared to the northern part because of the greater distance from sediment source areas in the north and because of lacustrine deposition in the ancient Lake Cahulla. In the southern Coachella Valley, the aquifer system consists of a semiperched zone that is fairly persistent southeast of Indio, an upper aquifer, a confining layer, and a lower aquifer. The general direction of groundwater flow is southeastward toward the Salton Sea." Text quoted directly from Sneed et al. (2014).	Sneed, M., Brandt, J.T., Solt, M. (2014). Land subsidence, groundwater levels, and geology in the Coachella Valley, California, 1993-2010. U.S. Geological Survey, Scientific Investigations Report 2014-5075, 62 pp. Accessed February 15, 2024 from https://pubs.usgs.gov/sir/2014/5075/pdf/sir2014-5075.pdf
Cuyama Valley	-	Thick alluvium and poorly/semi-consolidated sedimentary formations	Fig. 17A,B and Fig. 3 of Sweetkind et al. (2013): "The shallow alluvial section is subdivided into three units: Qc, fluvial channel deposits associated with the Cuyama River; Qya, younger alluvium; and Qoa, older alluvium (figs. 3 and 4; table 1). Previous studies did not separate the younger and older alluvium as separate units (Upson and Wortz, 1954; Singer and Swarzenski, 1970), but the two are distinguishable as mappable units at the surface (fig. 4) and, in the subsurface, can often be identified by differences in electric log signature. Younger alluvium (Qya, figs. 3	Sweetkind, D. S., Faunt, C. C., Hanson, R. T. (2013). Construction of 3-D geologic framework and textural models for Cuyama Valley groundwater basin, California. US Geological Survey Scientific Investigations Report 2013-5127, 58 pp. Accessed March 7, 2024 from https://pubs.usgs.gov/sir/2013/5127/pdf/sir2013-5127.pdf

Aquifer system title	Broader aquifer system title	Classification*	Basis for classification (cited figures are geologic cross-sections in most cases—text often quoted directly from reference in adjacent column (i.e., the column to the right))	Reference(s)
			and 4) consists of unconsolidated sand, gravel, and boulders, with some clay, deposited as alluvium in stream channels, floodplains, alluvial fans, and stream terraces. The unit is mainly Holocene in age, but locally can be late Pleistocene in part. Active stream deposits (Qc, figs. 3 and 4) consist of river-bed gravels of the Cuyama River and other active channels (Vedder and Repenning, 1975; DeLong and others, 2008, 2011). These deposits are incorporated into Qya within the geologic framework model." "Older alluvium (Qoa, figs. 3 and 4) consists of unconsolidated to partly consolidated sand, gravel, and boulders, with some clay, and the percentage of clay increases in the western part of the valley (Singer and Swarzenski, 1970; Vedder and Repenning, 1975; DeLong and others, 2008). Interpretation of geophysical logs from oil-exploration wells indicates that this unit is typically 125 to 200 m thick, but as thick as 300 m near the axis of Cuyama Valley (Schwing, 1984; Spitz, 1986). In the study area, older alluvium includes dissected alluvial fans, colluvial deposits, and sediments on multiple terraces and alluvial surfaces (Hill and others, 1958; DeLong and others, 2008). Older alluvium is exposed on uplifted alluvial surfaces along the south side of Cuyama Valley and in the center of the valley along the Turkey Trap and Graveyard Ridge faults (fig. 4; Vedder and Repenning, 1975; DeLong and others, 2008)." Text quoted directly from Sweetkind et al. (2013).	
Denver Basin	-	Consolidated elastic sedimentary-rock formations	Fig. 3 of Malenda and Penn (2020): "The bedrock aquifers in the Denver Basin aquifer system have a synorogenic, bowl-like structure composed of Late Cretaceous to Tertiary-age sandstone bedrock separated by unnamed claystone confining units (Fenneman, 1934; Robson, 1987; Paschke, 2011) and bounded at the base by the Cretaceous-age Pierre Shale (fig. 3, table 1). The four principal bedrock aquifers, from youngest (shallowest) to oldest (deepest), are the Dawson aquifer in the Dawson Formation, Denver aquifer in the Denver Formation, Arapahoe aquifer in the Arapahoe Formation, and Laramie-Fox Hills aquifer in the Laramie Formation and Fox Hills Sandstone (fig. 3). The consolidated sediments comprising these aquifers were deposited during different periods of mountain building and have physical properties (sediment grain size, porosity, specific yield, and hydraulic conductivity) that differ among aquifers and across the basin. Parts of the basin are overlain by unconsolidated alluvial aquifers (fig. 2)." Text quoted directly from Malenda and Penn (2020).	Malenda, H.F., Penn, C.A., (2020). Groundwater levels in the Denver Basin bedrock aquifers of Douglas County, Colorado, 2011–19. U.S. Geological Survey Scientific Investigations Report 2020–5076, 44 pp., Accessed November 29, 2021 from https://pubs.usgs.gov/sir/2020/5076/sir20205076.pdf

Aquifer system title	Broader aquifer system title	Classification*	Basis for classification (cited figures are geologic cross-sections in most cases—text often quoted directly from reference in adjacent column (i.e., the column to the right))	Reference(s)
Eastern Dakota Aquifer	-	Glacial drift overlying consolidated clastic sedimentary sequences	Figure on page 27 by Prior et al. (2003; see also table of aquifers and confining layers on pages 14–16 by Prior (2003)) where "Unconsolidated Quaternary deposits" are shown to be up to ~400 meters thick (further, the median alluvial thickness from Pelletier et al. (2016) is 48 m): "The Dakota aquifer is composed of sandstone deposits 200 to 300 feet in thickness, and provides water for many rural and public water supplies in northwest and west-central Iowa..." Text quoted from Prior et al. (2003).	Prior, J.C., Boekhoff, J.L., Howes, M.R., Libra, R.D., VanDorpe, P.E. (2003). Iowa's Groundwater Basics. Iowa Department of Natural Resources Report, 92 pp. Accessed November 29, 2021 from https://s-ihr34.ihr.uiowa.edu/publications/uploads/2014-08-24_08-08-24_es-06.pdf
Eureka and Eel River and Mad River Plains	-	Thick alluvium and poorly/semi-consolidated sedimentary formations	Fig. 3 of Johnson (1975): "Terrace deposits form step-like surfaces along the major river valleys and along streams of the Eureka Plain. The maximum thickness of the terrace deposits is about 100 ft. They yield water to wells locally from bodies of perched ground water and areally from deposits in the principal zone of saturation. In scattered localities, contact springs occur where saturated terrace deposits overlie relatively impermeable older formations. Alluvium—Alluvium underlies the flood plains of the major streams. Most of the irrigated agricultural land is alluvium. Much of the alluvium is poorly sorted sand and gravel deposits. Beneath the Eel River flood plain it is as much as 200 ft thick; most wells penetrate less than 70 ft. Beneath the Mad River flood plain the alluvium is as much as 100 ft thick, with many of the wells less than 30 ft deep. The deposits are unconsolidated and yield water readily to wells. Wells in the alluvium are the most productive in the area and have specific capacities ranging from 20 to 350 (gal/min)/ft. River channel deposits composed of coarse gravel and small amounts of coarse sand lie along the present channels of the Mad and Eel Rivers. Along the Mad River the deposits are upstream from the flood plain; along the Eel River they are in the channel and on the flood plain. Although the river channel deposits in the Eel River flood plain are saturated and highly permeable; they are tapped by few wells. Dune sand beach and dune sand occur in an almost continuous strip along the coast. The dune sand is more than 100 ft thick along the North Spit between the entrance to Humboldt Bay and the mouth of the Mad River. On this spit the dune sand is locally an important aquifer. It is developed as a source of water supply for shallow wells or well points that are driven into the sand far enough to penetrate the lens of freshwater overlying the seawater." Text quoted from Johnson (1975).	Johnson, M. J. (1975). Ground water conditions in the Eureka Area, Humboldt County, California. U.S. Geological Survey Water Resources Investigations 78-127. 61 pp. Accessed March 20, 2021 from https://pubs.usgs.gov/wri/1978/0127/report.pdf
Garber-Wellington Aquifer	-	Consolidated clastic sedimentary rock formations	Fig. 5 and Fig. 38 of Mashburn et al (2014); "Alluvium and terrace deposits in the study area of Quaternary age can be found along all major streams in the study area. These deposits are composed	Mashburn, S.L., Ryter, D.W., Neel, C.R., Smith, S.J., and Correll, J.S. (2014). Hydrogeology and simulation of ground water flow in the Central Oklahoma (Garber-Wellington) Aquifer, Oklahoma,

Aquifer system title	Broader aquifer system title	Classification*	Basis for classification (cited figures are geologic cross-sections in most cases—text often quoted directly from reference in adjacent column (i.e., the column to the right))	Reference(s)
			of lenticular beds of unconsolidated or loosely consolidated clays, silt, sand, and gravel with thicknesses ranging from 0 to 100 feet (ft) and "Beneath or adjoining the alluvium and terrace deposits are consolidated bedrock units of Permian age. The Permian-age geologic units in the Central Oklahoma aquifer study area dip gently to the west at approximately 50 feet per mile (Bingham and Moore, 1975)," and "the Permian-age Garber Sandstone and Wellington Formation have similar lithologies and differentiating the two units in surface outcrops and in the subsurface by geophysical logs or core samples is difficult. Therefore, these two geologic units are treated as a single hydrogeologic unit in this report. The Garber Sandstone and Wellington Formation consist of cross-bedded, fine-grained sandstone with interbedded shale or mudstone." Text quoted from Mashburn (2014).	1987 to 2009, and simulation of available water in storage, 2010–2050. U.S. Geological Survey Scientific Investigations Report 2013–5219, 92 pp. Accessed May 14, 2021 from https://pubs.usgs.gov/sir/2013/5219/pdf/sir20135219_v2.0.pdf
Honey Lake Valley	-	Thick alluvium and poorly/semi-consolidated sedimentary formations	Fig. 2 of Handman et al. (1990): "Principal geologic units in Honey Lake Valley are granitic bedrock, volcanic rocks, and unconsolidated to semiconsolidated sediments. The relative ages, thicknesses, and hydrologic properties of the geologic units are summarized in table 1 and their distribution is shown on plate 2"; Table 1 states that unconsolidated deposits range in thickness from 0 to 700 ft and cross sections in Figs. 2 and 3 confirm this. Underlying the unconsolidated materials are "Pliocene sedimentary and pyroclastic deposits" and basalts (see Table 1). The cross sections suggest that thick alluvium and poorly/semi-consolidated sedimentary formations are the most common units in the uppermost ~100–300 m of the crust (rather than basalt flows), and thus the aquifer system was categorized as thick alluvium and poorly/semi-consolidated sedimentary formations rather than as a volcanic unit.	Handman, E. H., Londquist, C. J., Maurer, D.K. (1990). Ground-water resources of Honey Lake valley, Lassen County, California, and Washoe County, Nevada. Water Resources Investigations Report 90-4050, 119 pp. Accessed March 21, 2021 from https://pubs.usgs.gov/wri/1990/4050/report.pdf
Judith Basin	-	Consolidated clastic sedimentary rock formations	"In the Judith basin, the Swift and Kootenai Formations and the Madison Group are the most productive and widespread aquifers from which to obtain a reliable water supply. For the most part, water in the Swift and Kootenai aquifers and in the individual sandstone lenses of the Colorado Shale is under artesian pressure, except near their respective outcrops. Water under water-table conditions is also obtainable from the discontinuous alluvial and terrace deposits throughout the basin, although these supplies are prone to contamination. In addition, the productivity of the terrace deposits is hampered by the limited thickness of the unit coupled with a limited source of recharge." and "The Upper Jurassic Swift Formation, the topmost formation of the Ellis Group, is a medium- to coarse-grained brown to orange glauconitic	Levings, J. F. (1983). Hydrogeology and simulation of water flow in the Kootenai aquifer of the Judith basin, central Montana. U.S. Geological Survey Water Resources Investigations Report 83-4146, 44 pp. Accessed March 29, 2021 from https://pubs.usgs.gov/wri/1983/4146/report.pdf

Aquifer system title	Broader aquifer system title	Classification*	Basis for classification (cited figures are geologic cross-sections in most cases—text often quoted directly from reference in adjacent column (i.e., the column to the right))	Reference(s)
			marine sandstone with interbeds of shale." and "The Upper Jurassic Morrison Formation conformably overlies the Swift. It consists mainly of 50 to 300 feet of nonmarine variegated shale and siltstone, thin nodular limestone, white and brown lenticular sandstone, and black shale." and "The basal part of the Kootenai Formation consists of a thick crossbedded fine to coarse grained "salt and pepper" fluvial sandstone and chert pebble conglomerate with a few shale breaks." and "The middle part of the formation, the Second Cat Creek sandstone, contains fine to coarse grained brown and gray sandstone lenses interspersed with siltstones and shales similar to those in the upper part." and "The upper third of the Kootenai consists primarily of red, purple, lavender, brown, and gray shale and siltstone." and "The Lower and Upper Cretaceous Colorado Shale crops out in large areas of the Judith basin (fig. 3). It consists of 1,500 to 2,000 feet of primarily fissile, easily weathered, dark gray to black marine shale interspersed with sandstone and bentonite beds. Near the base of this unit is a very fine grained sandstone with thin yellow and brown laminae called the First Cat Creek sandstone. Several isolated very fine grained sandstone beds interbedded with shale occur elsewhere in the lower 850 feet of the Colorado Shale. The Telegraph Creek Formation consists of the Upper Cretaceous transitional beds between the underlying Colorado Shale and overlying Eagle Sandstone. It consists of about 160 feet of yellow weathering dark gray to dark brown shale and sandy shale. This unit is included with the Colorado Shale (fig. 3) because of the difficulty of identifying its basal contact. Pleistocene terrace deposits from 0 to 20 feet thick cover large areas of the basin." text quoted directly from Levings (1983). Geologic strata descriptions are provided on page 9 of Levings (1983).	
Long Island	-	Glacial drift overlying poorly/semi-consolidated dipping layered clastic sedimentary sequences	Plate 1 by Smolensky and Buxton (1990) Fig. 6 by Masterson et al. (2016); "nearly all the groundwater withdrawals are from the surficial (locally referred to as the upper glacial) and Magothy aquifers (figs. 15B and 16) that either include the water table or are not separated to any large extent by confining units from the aquifer including the water table" and "...less permeable glacial moraine sediments in north-central Long Island."	Smolensky, D.A., Buxton, H.T., Shernoff, P.K. (1990). Hydrologic framework of Long Island, New York. U.S. Geological Survey Hydrologic Atlas 709, 3 plates. Accessed April 1, 2024 from https://pubs.usgs.gov/ha/709/plate-1.pdf Masterson, J. P., Pope, J. P., Fiener, M. N., Monti Jr, J., Nardi, M. R., Finkelstein, J. S. (2016). Assessment of groundwater availability in the Northern Atlantic Coastal Plain aquifer system from Long Island, New York, to North Carolina (No. 1829). US Geological Survey Professional Paper 1829, 90 pp. Accessed November 29, 2024 from https://pubs.usgs.gov/pp/1829/pp1829.pdf

Aquifer system title	Broader aquifer system title	Classification*	Basis for classification (cited figures are geologic cross-sections in most cases—text often quoted directly from reference in adjacent column (i.e., the column to the right))	Reference(s)
Los Angeles Basin	-	Thick alluvium and poorly/semi-consolidated sedimentary formations	Fig. 4 of Reichard (2003): "They grouped aquifers into four systems: the Recent, Lakewood, Upper San Pedro, and Lower San Pedro aquifer systems (fig. 2). The aquifer systems were defined on the basis of "unconformities, lithology, depositional characteristics, geochemistry, and vertical water level differences." The Recent aquifer system is predominantly composed of deposits of Holocene age and includes the Gaspar aquifer (California Department of Water Resources, 1961). The Lakewood aquifer system is composed of upper Pleistocene deposits and includes the Gage (also referred to as the 200-foot sand) and Gardena aquifers. The Upper San Pedro aquifer system is composed of lower Pleistocene deposits and includes the Lynwood (also referred to as the 400-foot gravel) and Silverado aquifers. The Lower San Pedro aquifer system is composed of lower Pleistocene deposits and includes the Sunnyside aquifer (also referred to as the Lower San Pedro aquifer). The Pico unit is considered as a low transmissive zone that underlies the Lower San Pedro aquifer system. The Pico unit is composed primarily of upper Pliocene to lower Pleistocene deposits." Text quoted directly from Land et al. (2016).	Reichard, E. G., Land, M., Crawford, S.M., Johnson, T.D., Everett, R.R., Kulshan, T.V., Ponti, D.J., Halford, K.L., Johnson, T.A., Paybins, K.S., Nishikawa, T. (2003). Geohydrology, Geochemistry, and Ground-Water Simulation-Optimization of the Central and West Coast Basins, Los Angeles County, California. USGS Numbered Series, 2003-4065. Pubs.er.usgs.gov, http://pubs.er.usgs.gov/publication/wri034065 Land, M., Reichard, E.G., Crawford, S.M., Everett, R.R., Newhouse, M.W., Williams, C.F. (2004). Ground-water Quality of Coastal Aquifer Systems in the West Coast Basin, Los Angeles County, California, 1999-2002. U.S. Geological Survey Scientific Investigations Report 2004-5067, 88 pp. Accessed February 15, 2021 from https://pubs.usgs.gov/sir/2004/5067/sir2004-5067.pdf
Michigan Basin	-	Glacial drift overlying consolidated clastic sedimentary sequences	Figs. 6 and 7 by Westjohn and Weaver (1998) shows thick (>100 m) glacial drift in much of the area: "Geologic units in the Michigan Basin regional aquifer system consist of Early Mississippian through Jurassic bedrock units and unconsolidated Pleistocene glacial deposits." Text quoted directly from Westjohn and Weaver (1998). The presence of moderately thick (tens of meters) and even thick (>100 m) glacial drift in much of the study area (especially the northwest) is supported by Fig. 30 by Olcott (1992).	Westjohn, D.B., Weaver, T.L. (1998). Hydrogeologic framework of the Michigan Basin regional aquifer system. US Geological Survey Professional Paper 1418, 55 pp. Accessed November 29, 2021 from https://pubs.usgs.gov/pp/1418/report.pdf Olcott, P.C. (1992). Groundwater Atlas of the United States: Segment 9 Iowa, Michigan, Minnesota, Wisconsin. U.S. Geological Survey Hydrologic Atlas 730-J, 33 pp. Accessed April 12, 2021 from https://pubs.usgs.gov/ha/730j/report.pdf
Mojave Basin	-	Thick alluvium and poorly/semi-consolidated sedimentary formations	"The calculated thickness of the basin-fill deposits, or depth to basement complex, ranged from 0 ft in the San Gabriel Mountains along the San Andreas Fault Zone and exposed buttes in the study area (shown as less than 250 ft on fig. 5) to about 5,775 ft in the northwestern part of the study area." Text quoted directly from Stamos et al. (2017).	Stamos, C. L., Christensen, A. H., Langenheim, V. (2017). Preliminary hydrogeologic assessment near the boundary of the Antelope Valley and El Mirage Valley groundwater basins, California. U.S. Geological Survey Scientific Investigations Report 2017-5065, 56 pp. Accessed March 20, 2021 from https://pubs.usgs.gov/sir/2017/5065/sir20175065.pdf
Northern Green River Basin	-	Consolidated clastic sedimentary rock formations	Fig. 7 of Bartos et al. (2015): "Rocks in the Bridger, Green River, Wasatch, and Fort Union Formations containing these aquifers occur at, near, or below the surface in the basin. Because fluvial sediments (primarily sandstone) compose most of the permeable rocks (aquifers), hydraulic properties generally depend on the	Bartos, T.T., Hallberg, L.L., and Eddy Miller, C.A., (2015). Hydrogeology, groundwater levels, and generalized potentiometric surface map of the Green River Basin lower Tertiary aquifer system, 2010-14, in the northern Green River structural basin, Wyoming. U.S. Geological Survey

Aquifer system title	Broader aquifer system title	Classification*	Basis for classification (cited figures are geologic cross-sections in most cases—text often quoted directly from reference in adjacent column (i.e., the column to the right))	Reference(s)
			number, thickness, and lateral and vertical continuity of sandstone layers and lenses.” and “Locally, fine-grained rocks with small primary permeability may yield water to wells where fractured or solution channels have developed, most notably fine-grained lacustrine rocks in the Laney Member of the Green River Formation (Laney aquifer).”	Scientific Investigations Report 2015–5090, 33 p., http://dx.doi.org/10.3133/sir20155090 . Accessed on May 13, 2024 from https://pubs.uegs.gov/sir/2015/5090/sir20155090.pdf
Ozark Plateaus Aquifer System	-	Carbonate rocks interbedded with clastic sedimentary rocks	Fig. 3 of Clark (2019): “The Ozark system is characterized by uplifted plateaus composed of relatively flat-lying sedimentary rocks of Paleozoic age that drape over basement rocks of Precambrian age. The hydrogeologic framework consists of interbedded carbonate and clastic units ranging in age from Cambrian to Early Pennsylvanian (Hays and others, 2016; Jorgensen and others, 1996; Westerman and others, 2016a) (fig. 2). Sandstone and karstified limestone and dolostone units serve as primary aquifers, and shale or dense dolostone generally act as confining units. In ascending order, the units include the Basement confining unit, St. Francois aquifer, St. Francois confining unit, lower Ozark aquifer, middle Ozark aquifer, upper Ozark aquifer, Ozark confining unit, Springfield Plateau aquifer, and Western Interior Plains confining system.” Text quoted from Clark et al. (2019).	Clark, B. R., Duncan, L. L., Krierim, K. J. (2019). Groundwater availability in the Ozark Plateaus aquifer system (No. 1854). US Geological Survey Professional Paper 1854, 95 pp. Accessed February 13, 2024 from https://pubs.er.usgs.gov/publication/pp1854
Pearl and Chattahoochee Aquifer System	-	Poorly-sorted, consolidated dipping layered-clastic sedimentary sequences	Figs. 75 of Miller (1990): “The regional aquifers are mostly sand with minor gravel and limestone beds, but they locally may contain clay beds. The regional confining units are primarily clay, silt, or chalk, but locally may contain sand beds. Each of the regional aquifers of the Southeastern Coastal Plain aquifer system has been named for a major river that crosses the outcrop belt of the aquifer and, thus, exposes the aquifer. From youngest to oldest, the four regional aquifers differentiated in the system are the Chickasawhay River aquifer, the Pearl River aquifer, the Chattahoochee River aquifer, and the Black Warrior River aquifer. The regional confining units separating these aquifers bear the same name of the aquifer they overlie. For example, the Pearl River regional confining unit lies above the Pearl River regional aquifer and beneath the Chickasawhay River regional aquifer, and so on (fig. 72).” Text quoted from Miller (1990).	Miller, J.A. (1990). Ground Water Atlas of the United States: Segment 6, Alabama, Florida, Georgia, South Carolina. U.S. Geological Survey Hydrologic Atlas 730-G, 30 pp. Accessed May 14, 2024 from https://www.nre.gov/docs/ML1706/ML17060B027.pdf
Peedee and Black-Creek and Cape Fear Aquifers	-	Poorly-sorted, consolidated dipping layered-clastic sedimentary sequences	Plates 5 and 13 by Winner Jr. and Coble (1989; see https://pubs.usgs.gov/of/1987/0690/plate-13.pdf): “The Coastal Plain sediments are characterized by: (1) mostly clastic rocks ranging from clay to gravel with lesser amounts of marine limestone, all resting on a foundation of crystalline basement	Winner Jr., M.D., Coble, R.W. (1989). Hydrogeologic framework of the North Carolina Coastal Plain aquifer system. U.S. Geological Survey Report 87-690, 167 pp. Accessed April 1, 2021 from https://pubs.usgs.gov/of/1987/0690/report.pdf

Aquifer system title	Broader aquifer system title	Classification*	Basis for classification (cited figures are geologic cross-sections in most cases—text often quoted directly from reference in adjacent column (i.e., the column to the right))	Reference(s)
			rocks; (2) a generally eastward dip; (3) a general thickening of beds toward the east; and, (4) the number of individual beds increasing in the seaward (eastward) direction." and "The Peedee aquifer is largely composed of the Peedee Sand of Late Cretaceous age..." and "The Peedee aquifer consists of nearly 70 percent sand, on the average, throughout the Coastal Plain" and "The North Carolina Coastal Plain is underlain by a generally eastward-dipping and eastward-thickening wedge of sedimentary rocks ranging in age from Holocene to Cretaceous and composed of unconsolidated gravel, sand, silt, and clay with scattered beds of shells, indurated to loosely consolidated beds of limestone, sandy limestone, and shell limestone" and "The Black Creek Formation is lagoonal to marine, consisting of thinly laminated gray to black clay interlayered with gray to tan sands. In some outcrops, the formation is characterized by sand-dominated or clay-dominated lenses. Other outcrops show well-defined beds of clean sand and gray to black clay. A primary characteristic of Black Creek sediments in the subsurface is their high content of organic material, particularly lignitized wood. Shell material and glauconite are also common." and "The outcropping Cape Fear consists of alternating beds of sand and clay that are commonly 3 to 5 feet thick but range from a fraction of a foot to 15 feet in thickness. Some beds show vertical gradation from sand to clay, while others carry thin conglomerates of quartz pebbles or mudstone fragments." Text quoted directly from Winner Jr. and Coble (1989). Surficial aquifer is shown to be ~50 ft. thick in Plates 5 and 13 by Winner Jr. and Coble (1989).	
Salinas Valley	-	Thick alluvium and poorly/semi-consolidated sedimentary formations	"The Salinas Valley groundwater basin (Figure 1) consists of four subareas (Pressure, East Side, Forebay, and Upper Valley) and two main water-bearing zones, the upper "180-foot" (55 m) and "400-foot" (122 m) aquifers which are separated by an impermeable clay layer at ~80 m that is up to 30 m thick (Figure 2). A perched aquifer overlies the 180-foot aquifer in the northern part of the valley above a clay layer that is usually, but not uniformly, present at a depth \geq 15 m (the Salinas Valley Aquiclude). A deep lower aquifer system underlies the 400-foot aquifer. The 180-foot aquifer is up to 55 m thick. The perched and 180-foot aquifers consists of Pleistocene to Holocene gravels, sands, silts, and clays, which were deposited as alluvial/fluvial valley fill. The 400-foot aquifer lies at a depth of 82–143 m and is part of the Pleistocene Aromas and Pliocene Paso Robles formations. The aquifer consists of poorly bedded sands, gravels, volcanic tuff, and interbedded clays, which act as local aquitards. [1970] showed that both the 180- and 400-foot aquifers crop out	Vengosh, A., Gill, J., Lee-Davison, M., & Bryant-Hudson, C. (2002). A multi-isotope (B, Sr, O, H, and C) and age-dating (^3H , ^3He and ^{14}C) study of groundwater from Salinas Valley, California: Hydrochemistry, dynamics, and contamination processes. Water Resources Research , 38, 1008.

Aquifer system title	Broader aquifer system title	Classification*	Basis for classification (cited figures are geologic cross-sections in most cases—text often quoted directly from reference in adjacent column (i.e., the column to the right))	Reference(s)
			on the walls of Monterey Bay submarine canyon, which allows direct hydraulic connection between fresh and ocean water." Text quoted directly from Vengosh et al. (2002).	
Salt Lake Valley	-	Thick alluvium and poorly/semi-consolidated sedimentary formations	"The basin-fill deposits in the valley consist of unconsolidated to semiconsolidated Tertiary-age sediments overlain by unconsolidated Quaternary-age sediments. The less permeable Tertiary-age basin-fill deposits are associated with alluvial fans and volcanic ash. The saturated Quaternary-age basin-fill deposits range from less than 200 ft thick in the Kearns area and along the margins of the valley to more than 1,000 ft thick in the northern part of the valley." Text quoted from Thiroe (2003).	Thiroe, S.A. (2003). Hydrogeology of shallow basin-fill deposits in areas of Salt Lake Valley, Salt Lake County, Utah. U.S. Geological Survey Water Resources Investigations Report 2003-4029, 32 pp. Accessed March 27, 2021 from https://pubs.usgs.gov/wri/wri034029/pdf/wri034029.pdf
San Antonio Creek Valley	-	Thick alluvium and poorly/semi-consolidated sedimentary formations	Fig. 2A,B of Martin (1984): "The lithology of the sediments has been generalized by Muir (1964) into two categories: consolidated non-water-bearing rocks and unconsolidated water-bearing deposits. The consolidated rocks are sedimentary rocks, predominantly marine in origin, that are nearly impermeable except for slightly permeable sandstones and for fracture zones. The consolidated rocks form the lower boundary of the aquifer and also form much of the perimeter boundary of the ground-water basin (fig. 1). Unconsolidated deposits of predominantly sand and gravel blanket the central part of the valley and form the ground-water basin. In upward succession, the unconsolidated deposits include the lower and upper members of the Careaga Sand of Pliocene age, the Paso Robles Formation of Pliocene and Pleistocene age, the Orcutt Sand of Pleistocene age, the terrace deposits of Pleistocene age, and the alluvium of Holocene age. The relative placement of these deposits is shown diagrammatically in figure 2. The unconsolidated deposits range in thickness from zero feet along the perimeter of the basin to about 3,000 feet in the central part of the basin near Los Alamos (Hutchinson, 1980, p. 15)." Text quoted directly from Martin (1984).	Martin, P. (1984). Development and calibration of a two-dimensional digital model for the analysis of the ground-water flow system in the San Antonio Creek Valley, Santa Barbara County, California. Water Resources Investigations Report 84-4340, 73 pp. Accessed March 7, 2021 from https://pubs.usgs.gov/wri/wri844340/report.pdf
San Pedro Basin	-	Thick alluvium and poorly/semi-consolidated sedimentary formations	Fig. 5.2 of Callegary (2016): "The sedimentary fill within the U.S. portion of the BSPB has been divided into two informal units, called Lower Basin Fill and Upper Basin Fill (Brown et al., 1966), which were deposited in structural basins between the mountains during the late Miocene-Plio-Pleistocene. Although in the Mexican portion of the basin there are no detailed studies on the stratigraphy of these sediments which also represent the primary regional alluvial aquifer, the physical characteristics obtained from the lithologic descriptions of wells (Consultores en Agua	Callegary, J.B., Minjárez-Sosa, I., Tapia-Villaseñor, E.M., dos Santos, P., Monreal-Saavedra, R., Grijalva-Noriega, F.J., Huth, A.K., Gray, F., Scott, C.A., Megdal, S.B., Oroz-Ramos, L.A., Rangel-Medina, M., Leenhouts, J.M. (2016). San Pedro River Aquifer Binational Report: International Boundary and Water Commission, 170 pp. Accessed November 29, 2021 from https://ibwc.gov/Files/San_Pedro_River_Binational%20Report_013146.pdf

Aquifer system title	Broader aquifer system title	Classification*	Basis for classification (cited figures are geologic cross-sections in most cases—text often quoted directly from reference in adjacent column (i.e., the column to the right))	Reference(s)
			Subterránea, S.A., 2000) suggest an equivalence with the division presented in the United States. With data obtained from the lithological description of wells, vertical electrical soundings, and other geophysical methods (Pool and Coes, 1999; Consultores en Agua Subterránea, S.A., 2000; Fleming and Pool, 2002; Conder Consulting, 2001, 2003), Pool and Dickinson (2007) constructed continuous cross-border surfaces delineating the top and bottom of the silty-clay zone within the Upper Basin Fill. They estimated it to be from 10 to 300 m thick, and laterally confined to within a few kilometers of the San Pedro River channel. Based on these data, the elevations of this zone vary between 1400 and 1100 m.a.s.l." Text quoted directly from Callegary et al. (2016).	
Santa Clara-Calleguas Basin	-	Thick alluvium and poorly/semi-consolidated sedimentary formations	Fig. 8 of Hanson (2002): "The Quaternary unconsolidated deposits consist of the Santa Barbara Formation (Weber and others, 1976), the Las Posas Sand (Dibblee, 1988, 1990a,b, 1991, 1992a,b,c,d; Dibblee and Ehrenspeck, 1990), the San Pedro Formation (Weber and others, 1976), and the Saugus Formation (Weber and others, 1976; Dibblee, 1988, 1990a,b, 1991, 1992 a,b,c,d), all of the Pleistocene epoch, and unconsolidated alluvial and fluvial deposits of the Pleistocene to Holocene epoch. In the Santa Clara Calleguas Basin, the unconsolidated deposits are grouped together into the upper aquifer system and the lower aquifer system" and "The upper part of San Pedro Formation consists of lenticular layers of sand, gravel, silt, and clay of marine and continental origin. The continental fluvial silt, sand, and gravel deposits within the upper part of the San Pedro Formation are referred to as the Saugus Formation by Dibblee (1988, 1990a,b, 1991, 1992a,b,c,d) and Dibblee and Ehrenspeck (1990). These deposits reach a maximum thickness of more than 5,000 ft in the Piru subbasin in the Santa Clara River Valley (Dibblee, 1991). The sand and gravel layers range from 10 to 100 ft thick and are separated by silt and clay layers that generally are 10 to 20 ft thick." Text quoted directly from Hanson et al. (2002).	Hanson, R.T., Martin, P., Koczot, K.M. (2002). Simulation of ground-water/surface-water flow in the Santa Clara-Calleguas ground-water basin, Ventura County, California. Water Resources Investigations Report 2002-4136, 172 pp. Accessed March 20, 2021 from https://pubs.usgs.gov/wri/wri024136/wri024136.pdf
Santa Rosa Valley	-	Thick alluvium and poorly/semi-consolidated sedimentary formations	Plates 3 and 4 of Cardwell (1958): "Santa Rosa and Petaluma Valleys are the westernmost of the several small valleys immediately north of San Francisco Bay, California. The two valleys occupy aligned, structurally controlled depressions in the Coast Ranges of northern California. Together they extend from the northern margin of San Francisco Bay northwestward about 35 miles to the Russian River. The valleys are underlain by unconsolidated marine and continental sediments and volcanic rocks of Tertiary and Quaternary age. This material is water	Cardwell, G. T., (1958). Geology and ground water in the Santa Rosa and Petaluma Valley areas, Sonoma County, California. U.S. Geological Survey Water Supply Paper 1427, 284 pp. Accessed November 20, 2021 from https://pubs.usgs.gov/wsp/1427/report.pdf

Aquifer system title	Broader aquifer system title	Classification*	Basis for classification (cited figures are geologic cross-sections in most cases — text often quoted directly from reference in adjacent column (i.e., the column to the right))	Reference(s)
			bearing in large part and makes up a relatively deep ground-water basin, Santa Rosa Valley, the northernmost and larger of the two valleys, contains about 90 square miles of the approximately 150 square miles of plains and essentially flat-lying lands in the area." Text quoted directly from Cardwell (1958).	
South Park Basin	-	Consolidated sedimentary rocks (elastic and carbonate)	Hydrostratigraphy in Fig. 2a by Donegan (2018) Fig. 11 by Barkmann et al. (2013); cross sections highlight that portions of the basin are underlain by carbonate rocks (see Paleozoic Sedimentary Units in Fig. 11 by Barkmann et al. (2013) and other areas are underlain by clastic sedimentary rocks (e.g., see Laramide sedimentary units or Mesozoic sedimentary units and their distributions depicted in cross sections in Fig. 11 by Barkmann et al. (2013)). The aquifer system was thus classified broadly as "Consolidated sedimentary rocks" as both clastic and carbonate rocks are widespread in the area.	Donegan, K.C. (2018). Groundwater levels in the South Park Basin 2018. Colorado Division of Water Resources Report, 29 pp. Accessed November 29, 2021 from https://dnrweblink.state.co.us/dwr/ElectronicFile.aspx?docid=3351305&dbid=0 Barkmann, P.E., Moore, A., Johnson, J. (2013). South Park Groundwater Quality Scoping Study. Report for Coalition for the Upper South Platte, 74 pp. Accessed November 29, 2021 from https://cusp.ws/wp-content/uploads/2014/10/South-Park-Groundwater-Quality-Scoping-Study_Final.pdf
Southern San Juan Basin	-	Consolidated clastic sedimentary rock formations	Fig. 6 of Kernodle (1996): "Quaternary and recent deposits in the San Juan Basin include stream-deposited alluvium and older terrace deposits, landslide deposits, and eolian sand. The areal distribution of these sediments are shown in figure 12. Most Quaternary and younger deposits are unconsolidated and form a thin covering over older bedrock sediments." and "The San Juan structural basin contains a thick sequence of sedimentary rocks ranging in age from Cambrian through Tertiary (fig. 5), but principally from Pennsylvanian through Tertiary. The maximum thickness of the sequence of rocks is about 14,000 feet (Fassett and Hinds, 1971, p. 4). These sedimentary rocks dip basinward from the basin margins toward the troughlike structural center (deepest part of the basin) except where locally interrupted by intrabasin folds, faults, and domes. Older sedimentary rocks crop out around the basin margins and are successively overlain by younger rocks toward the center of the structural basin (fig. 6). Volcanic rocks of Tertiary age and various deposits of Quaternary age also are present in the basin." Text quoted directly from Kernodle (1996).	Kernodle, J. M. (1996). Hydrogeology and steady-state simulation of ground-water flow in the San Juan Basin, New Mexico, Colorado, Arizona, and Utah. US Geological Survey Water Resources Investigations Report 95-4187, 126 pp. Accessed March 12, 2021 from https://pubs.usgs.gov/wri/1995/4187/report.pdf
Spanish Springs Valley	-	Thick alluvium and poorly/semi-consolidated sedimentary formations	Figs. 7-9 of Berger et al. (1997): "The structural depression occupied by Spanish Springs Valley is partly filled by interbedded deposits of sand, gravel, clay, and silt derived primarily from adjacent mountains. These deposits form the basin-fill aquifer, which is bounded and underlain by consolidated rock." and "Wells drilled in Spanish Springs Valley range in depth from several tens	Berger, D. L., Ross, W. C., Thodal, C. E., Robledo, A. R. (1997). Hydrogeology and simulated effects of urban development on water resources of Spanish Springs Valley, Washoe County, West-Central Nevada. U.S. Geological Survey Water Resources Investigations Report 96-4297, 87 pp. Accessed March 22, 2021 from https://pubs.usgs.gov/wri/1996/4297/report.pdf

Aquifer system title	Broader aquifer system title	Classification ^a	Basis for classification (cited figures are geologic cross-sections in most cases — text often quoted directly from reference in adjacent column (i.e., the column to the right))	Reference(s)
			of feet to more than 800 ft, with most completed in basin fill. Discrepancies in basin fill thickness as reported on drillers' logs for several wells limited the areal use of these logs as a tool to estimate basin fill thickness. The basin fill thickness map (fig-10), which was developed partly by geophysical methods, indicates that basin fill is thickest along a northeast-trending trough-like feature close to the mountain front of Hungry Ridge." Text quoted from Berger et al. (1997).	
Tijuana-San Diego	-	Thick alluvium and poorly/semi-consolidated sedimentary formations	"Quaternary Alluvium. The most productive unit in the basin is the alluvium which consists of river and stream deposits of gravel, sand, silt, and clay. The thickness of the alluvium is less than 150 feet (Izbicki 1985) and averages about 80 feet thick (SDCWA 1997). Wells yield as much as 2,000 gpm, but the average yield is 1,000 gpm (SDCWA 1997). Groundwater in this unit is unconfined and the specific yield is about 15 percent (SDCWA 1997). San Diego Formation. This unit consists of Pliocene age well-sorted, medium to coarse sand, silty and clayey sand, sandy silt, and sandy clay (Huntley and others 1996). Thickness of this unit ranges to at least 1,700 feet in the basin (Dudek and Associates 1994). Well yields average about 350 gpm with discharges as high as 1,000 gpm being recorded. Groundwater in the San Diego Formation is confined with a storage coefficient of about 0.004 (SDCWA 1997)" text quoted directly from California Department of Water Resources (2003).	California Department of Water Resources (2003). California's Groundwater Bulletin 118, Hydrologic Region South Coast; Tijuana Groundwater Basin, 3 p. Accessed November 29, 2021 from https://water.ca.gov/-/media/DWR-Website/Web-Pages/Programs/Groundwater-Management/Bulletin-118/Files/2003-Basin-Descriptions/9_019_Tijuana.pdf
Upper Santa Ana Basin	-	Thick alluvium and poorly/semi-consolidated sedimentary formations	"Aquifers of the Upper Santa Ana Valley Groundwater Basin are generally unconfined and comprise several subbasins filled with alluvial deposits eroded from the surrounding mountains (Hamlin and others, 2005). The thickness of these deposits ranges from less than 200 ft to more than 1,000 ft (Dutcher and Garrett, 1963). Faults play an important role in the ground water flow system here." Text quoted directly from Kent and Belitz (2009).	Kent, R., Belitz, K. (2009). Ground water quality data in the Upper Santa Ana Watershed Study Unit, November 2006 to March 2007: Results from the California GAMA Program. U.S. Geological Survey Data Series 404, 116 pp. Accessed March 21, 2021 from https://pubs.usgs.gov/ds/404/ds404.pdf
Utah Lake Valley	-	Thick alluvium and poorly/semi-consolidated sedimentary formations	"It has been assumed until only very recently that the depth of fill approached a probable limit of some 8000 to 10,000 feet (2400 to 3000 meters). However, information from the drilling of a wildcat well about one mile (1.6 kilometers) west of the city of Spanish Fork by the Gulf Oil Company during the fall of 1977 proved that limit very conservative. While much of the information is still confidential, a company spokesman did say that drilling was abandoned without reaching bedrock at a depth of 13,000 feet (3962 meters)." Text quoted directly from Dustin (1978).	Dustin, J.D. (1978). Hydrogeology of Utah Lake with Emphasis on Goshen Bay. PhD Dissertation, Brigham Young University, 184 pp. Accessed March 27, 2021 from https://apps.dtic.mil/sti/pdfs/ADA065478.pdf

^aclassifications based primarily on geologic conditions in the uppermost 100-300 m of Earth's crust; classifications are generalizations of complex sequences of sediment and bedrock that together make of the aquifer system

Supplementary Table 7. Median and lower upper quartile range of depths below which *most (i.e., 60%, 70% or 80%) wells contain minimal (<25%) modern groundwater among aquifer classifications

Classification	Count**	60%*	70%*	80%*
Carbonate rocks interbedded with clastic sedimentary rocks	6	71 (35-96)	113 (34-113)	163 (52-192)
Glacial drift overlying carbonate rocks	4	*	*	*
Unconsolidated deposits overlying carbonate rocks	6	8 (6-31)	40 (15-56)	59 (17-61)
Consolidated clastic sedimentary rock formations	8	30 (21-43)	48 (21-52)	60 (29-70)
Glacial drift overlying consolidated clastic sedimentary sequences	7	56 (10-68)	47 (17-83)	30 (28-103)
Glacial drift overlying poorly/semi-consolidated dipping layered clastic sedimentary sequences	1	*	*	*
Moderately thick unconsolidated aquifer overlying layered clastic sedimentary sequences	4	34 (29-48)	48 (37-69)	71 (59-88)
Poorly/semi-consolidated dipping layered clastic sedimentary sequences	16	15 (7-32)	24 (15-36)	29 (17-55)
Thick alluvium and poorly/semi-consolidated sedimentary formations	31	77 (30-124)	94 (64-189)	152 (89-224)
Volcanic (basalt) rocks and alluvial deposits	6	36 (31-95)	62 (33-115)	130 (78-139)
Consolidated sedimentary rocks (clastic and carbonate)	2	*	*	*

* depths below which most wells contain minimal (25%) modern groundwater becomes scarce in aquifer systems (median and lower upper quartile of depths for aquifers with a given classification); results presented for classifications with at least 5 aquifer systems (in Supplementary Table 5)

** number of unique study aquifers (for a given classification) with sufficient tritium data to determine the below which most (i.e., 60%, 70% or 80%) wells contain minimal (<25%) modern groundwater

Supplementary Note 4. Delineating aquifer boundaries by review of local studies

We approximated the boundaries of aquifer systems across the United States on the basis of an extensive literature review. Specifically, we reviewed peer-reviewed research and grey literature and identified maps and descriptions of physiographic settings treated as study areas by these local- and regional-scale studies and georeferenced figures (i.e., maps) from these studies to develop a geodatabase of aquifer system boundaries across the United States.

We are aware that a geodatabase of aquifers is available from the United States Geological Survey (https://water.usgs.gov/GIS/metadata/usgswrd/XML/aquifers_us.xml), however, we preferred to develop a new geodatabase of aquifer system outlines based on our own review of local- and regional-scale studies for numerous reasons, some of which include (i) that many of the polygons in the USGS' Principal Aquifers Database are too expansive for locally-relevant science (for example, the entire High Plains aquifer system is a single polygon in the USGS database, whereas regional-scale research in these areas separate the broader High Plains aquifer system into several subareas e.g., "Great Bend Prairie" and "Equus Beds" see <https://geokansas.ku.edu/kansas-high-plains-aquifer-atlas>), and (ii) some aquifers identified in local- and regional-scale studies are not named nor outlined in the USGS Principal Aquifers Database (e.g., the Milk River Aquifer System).

We mapped hundreds of aquifer systems across the USA; as the 2D boundaries of aquifer units are challenging to map these polygons are best treated as hydrogeologic study areas, but referred to herein as 'aquifer systems'. We then analysed depths that post-1953 groundwater is common in a subset of these aquifer systems for which sufficient well-water tritium data were available (see Methods in main text). For transparency we provide specific details about the aquifer system boundary delineation process and literature sources for all of the aquifer systems we delineated below.

Formatted: Normal

Formatted: Font: Times New Roman, English (United Kingdom)

Formatted: Strong

Supplementary Table 8. Steps applied (and primary literature sources consulted) to delineate hundreds of aquifer systems

Aquifer System	Broader System	References	Delineation steps and sources		
Milk River	-	Pétre, M. A., Rivera, A., Lefebvre, R., Hendry, M. J., Fohnagy, A. J. (2016). A unified hydrogeological conceptual model of the Milk River transboundary aquifer, traversing Alberta (Canada) and Montana (USA). Hydrogeology Journal , 24, 1847-1874.	Phillips, F. M., Bentley, H. W., Davie, S. N., Elmore, D., Swanick, G. B. (1986). Chlorine-36 dating of very old groundwater: 2. Milk River aquifer, Alberta, Canada. Water Resources Research , 22, 2003-2016.	-	Approximated from Fig. 2 of Pétre et al. (2016) and Figs. 1 and 2 by Phillips et al. (1986)
Oak Ridges Moraine	-	Gerber, R. E., Howard, K. (2002). Hydrogeology of the Oak Ridges Moraine aquifer system: implications for protection and management from the Duffins Creek watershed. Canadian Journal of Earth Sciences , 39(9), 1333-1348.	Marchildon, M., Kassenaar, D. (2013). Analyzing low impact development strategies using continuous fully distributed coupled groundwater and surface water models. Journal of Water Management Modeling , R246-17, http://dx.doi.org/10.14796/JWMM.R246-17 .	-	Approximated from Figs. 1 and 2 by Gerber and Howard (2002), and Fig. 17.4 by Marchildon and Kassenaar (2013)
Black Warrior River Aquifer System (Eutaw and McShan Formations and Tuscaloosa Group)	-	Kidd, R. E., Lambeth, D. S. (1995). Hydrogeology and ground-water quality in the Black Belt area of west-central Alabama, and estimated water use for aquaculture, 1990 (Vol. 94, No. 4074). US Department of the Interior, US Geological Survey. Accessed February 10, 2021 via https://citeseerx.ist.psu.edu/viewdoc/download?doi=10.1.1.1015.2227&rep=rep1&type=pdf	Strom, E. W., Mallory, M. J. (1995). Hydrogeology and simulation of ground water flow in the Eutaw-McShan Aquifer and in the Tuscaloosa aquifer system in northeastern Mississippi. US Geological Survey Water Resources Investigations Report 94-4223, 89 pp. Accessed November 29, 2021 from https://pubs.usgs.gov/wri/1994/4223/report.pdf	Miller, J. A. (1990). Ground Water Atlas of the United States: Segment 6. Alabama, Florida, Georgia, South Carolina. U.S. Geological Survey Hydrologic Atlas 730-G, 30 pp. Accessed April 5, 2024 from https://www.nrc.gov/docs/ML1706/ML17060B027.pdf	Mississippi portion approximated from Fig. 8 of Strom and Mallory (1995). Alabama portion approximated from Fig. 3 of Kidd and Lambeth (1995) and Fig. 73 by Miller (1990) for easternmost extent. Best estimates were made to connect these two separate delineations in the northwest portion of the aquifer's extent within Alabama.
Equus Beds	High Plains	Gutentag, E. D., Heimes, F. J., Krothe, N. C., Luckey, R. R., Weeks, J. B. (1984). Geohydrology of the High Plains aquifer in parts of Colorado, Kansas, Nebraska, New Mexico, Oklahoma, South Dakota, Texas, and Wyoming (No. 1400-B). Accessed February 10, 2021 from https://pubs.usgs.gov/pp/1400b/report.pdf	Kansas Geological Survey (2021). Webpage entitled "High Plains Aquifer Regions in Kansas" (Kansas High Plains Aquifer Atlas). Accessed April 15, 2021 from https://geokansas.ku.edu/kansas-high-plains-aquifer-atlas	-	Approximated from Fig. 1 of Gutentag et al. (1984). Great Bend Prairie area approximated from the "High Plains Aquifer Regions in Kansas" (Kansas High Plains Aquifer Atlas) accessed April 15, 2021 from the Kansas Geological Survey (2021); https://geokansas.ku.edu/kansas-high-plains-aquifer-atlas
Great Bend Prairie	High Plains	Gutentag, E. D., Heimes, F. J., Krothe, N. C., Luckey, R. R., Weeks, J. B. (1984). Geohydrology of the High Plains aquifer in parts of Colorado, Kansas, Nebraska, New Mexico, Oklahoma, South Dakota, Texas, and Wyoming (No. 1400-B). Accessed February 10, 2021	Kansas Geological Survey (2021). Webpage entitled "High Plains Aquifer Regions in Kansas" (Kansas High Plains Aquifer Atlas). Accessed April 15, 2021 from https://geokansas.ku.edu/kansas-high-plains-aquifer-atlas	-	Approximated from Fig. 1 of Gutentag et al. (1984). Great Bend Prairie area approximated from the "High Plains Aquifer Regions in Kansas" (Kansas High Plains Aquifer Atlas) accessed April 15, 2021 from the Kansas Geological

Aquifer System	Broader System	References	Delineation steps and sources		
		from https://pubs.usgs.gov/pp/1400b/report.pdf	Survey (2021; https://geokansas.ku.edu/kansas-high-plains-aquifer-atlas)		
Northern High Plains	High Plains	Peterson, S. M., Traylor, J. P., Guira, M. (2020). Groundwater Availability of the Northern High Plains Aquifer in Colorado, Kansas, Nebraska, South Dakota, and Wyoming. U.S. Geological Survey Professional Paper 1864, 57 p., Accessed February 10, 2021 from https://doi.org/10.3133/pp1864 .	Miller, J. A., Appel, C. L. (1997). Ground Water Atlas of the United States: Kansas, Missouri, and Nebraska HA. United States Geological Survey Report 730-D. Accessed February 10, 2021 from https://pubs.usgs.gov/hq/ha730/ch_d/ .	Gutentag, E. D., Heimes, F., Krothe, N. C., Luckey, R., Weeks, J. B. (1984). Geohydrology of the High Plains aquifer in parts of Colorado, Kansas, Nebraska, New Mexico, Oklahoma, South Dakota, Texas, and Wyoming (No. 1400-B). Accessed February 10, 2021 from https://pubs.usgs.gov/pp/1400b/report.pdf	Approximated from Fig. 7 of Peterson et al. (2020)
Central High Plains	High Plains	Gutentag, E. D., Heimes, F., Krothe, N. C., Luckey, R., Weeks, J. B. (1984). Geohydrology of the High Plains aquifer in parts of Colorado, Kansas, Nebraska, New Mexico, Oklahoma, South Dakota, Texas, and Wyoming (No. 1400-B). Accessed February 10, 2021 from https://pubs.usgs.gov/pp/1400b/report.pdf	-	-	Approximated from Fig. 4 of Gutentag et al. (1984)
Southern High Plains	High Plains	Gutentag, E. D., Heimes, F., Krothe, N. C., Luckey, R., Weeks, J. B. (1984). Geohydrology of the High Plains aquifer in parts of Colorado, Kansas, Nebraska, New Mexico, Oklahoma, South Dakota, Texas, and Wyoming (No. 1400-B). Accessed February 10, 2021 from https://pubs.usgs.gov/pp/1400b/report.pdf	-	-	Approximated from Fig. 4 of Gutentag et al. (1984)
Confined Claiborne Near Jackson	Mississippi Embayment	Stephenson, L. W. (1941). The ground-water resources of Mississippi. U.S. Geological Survey Water Supply Paper 676, 540 pp. Accessed March 11, 2021 from https://pubs.usgs.gov/wsp/0576/report.pdf	-	-	Discussion on page 200 describing access to Claiborne aquifer (part of the Mississippi Embayment Aquifer System) by wells near Jackson
Eastern Mississippi Embayment	Mississippi Embayment	Arthur, J. K., Taylor, R. E. (1998). Ground-water flow analysis of the Mississippi embayment aquifer system,	Renken, R. A. (1998). Groundwater Atlas of the United States: Arkansas, Louisiana, Mississippi (HA-730-F),	-	Approximated from Fig. 2 of Arthur and Taylor (1998) and Figs. 8-10 and Fig. 65 by Renken (1998)

Aquifer System	Broader System	References	Delineation steps and sources		
		South-Central United States. US Geological Survey Professional Paper 1416-1, 59 pp. Accessed February 10, 2021 from https://pubs.usgs.gov/pp/1416i/report.pdf	U.S. Geological Survey Hydrologic Investigations Atlas 730-F, 30 pp. Accessed November 28, 2021 from https://pubs.usgs.gov/ha/730f/report.pdf		
Central-Mississippi Embayment	Mississippi Embayment	Arthur, J. K., Taylor, R. E. (1998). Ground-water flow analysis of the Mississippi embayment aquifer system, South-Central United States. US Geological Survey Professional Paper 1416-1, 59 pp. Accessed February 10, 2021 from https://pubs.usgs.gov/pp/1416i/report.pdf	Renken, R. A. (1998). Groundwater Atlas of the United States: Arkansas, Louisiana, Mississippi (HA-730-F). U.S. Geological Survey Hydrologic Investigations Atlas 730-F, 30 pp. Accessed November 28, 2021 from https://pubs.usgs.gov/ha/730f/report.pdf	-	Approximated from Fig. 2 of Arthur and Taylor (1998) and Figs. 8-10 and Fig. 65 by Renken (1998)
Western-Mississippi Embayment	Mississippi Embayment	Arthur, J. K., Taylor, R. E. (1998). Ground-water flow analysis of the Mississippi embayment aquifer system, South-Central United States. US Geological Survey Professional Paper 1416-1, 59 pp. Accessed February 10, 2021 from https://pubs.usgs.gov/pp/1416i/report.pdf	Benken, R. A. (1998). Groundwater Atlas of the United States: Arkansas, Louisiana, Mississippi (HA-730-F). U.S. Geological Survey Hydrologic Investigations Atlas 730-F, 30 pp. Accessed November 28, 2021 from https://pubs.usgs.gov/ha/730f/report.pdf	-	Approximated from Fig. 2 of Arthur and Taylor (1998) and Figs. 8-10 and Fig. 65 by Renken (1998)
Eastern Carrizo-Wilcox	Carrizo-Wilcox	Huang, Y., Scanlon, B. R., Nicol, J. P., Reedy, R. C., Dutton, A. R., Kelley, V. A., Deeds, N. E. (2012). Sources of groundwater pumpage in a layered aquifer system in the Upper Gulf Coastal Plain, USA. Hydrogeology Journal, 20(4), 783-796.	Kelley, V. A., Deeds, N. E., Fryar, D. G., Nicol, J. P. (2004). Groundwater availability models for the Queen City and Sparta aquifers. Contract report to the Texas Water Development Board, Austin, TX, 867 pp.	Louisiana Department of Environmental Quality (2007). Carrizo-Wilcox Aquifer Summary Report 2007. Aquifer Sampling and Assessment Program (ASSET) Program. Accessed February 11, 2021 from https://deq.louisiana.gov/assets/does/Water/Triennial_reports/AquiferSummaries_2007-2009/02Carrizo-WilcoxAquiferSummary09.pdf	Texas portion approximated from Fig. 4 of Huang et al. (2012). Louisiana portion approximated from Fig. 2-4 of Louisiana Department of Environmental Quality (2007). Western boundary is the western margin of the Texas Groundwater Management Area (GMA) #11 (accessed February 11, 2021 from https://www.twdb.texas.gov/groundwater/management_areas/gma12.asp)
Central Carrizo-Wilcox	Carrizo-Wilcox	Huang, Y., Scanlon, B. R., Nicol, J. P., Reedy, R. C., Dutton, A. R., Kelley, V. A., Deeds, N. E. (2012). Sources of groundwater pumpage in a layered aquifer system in the Upper Gulf Coastal Plain, USA. Hydrogeology Journal, 20(4), 783-796.	Kelley, V. A., Deeds, N. E., Fryar, D. G., Nicol, J. P. (2004). Groundwater availability models for the Queen City and Sparta aquifers. Contract report to the Texas Water Development Board, Austin, TX, 867 pp.	-	Texas portion approximated from Fig. 4 of Huang et al. (2012). Eastern and western boundaries approximated by Texas' 12th Groundwater Management Area (GMA) (accessed February 11, 2021 from https://www.twdb.texas.gov/groundwater/management_areas/gma12.asp)
Western Carrizo-Wilcox	Carrizo-Wilcox	Huang, Y., Scanlon, B. R., Nicol, J. P., Reedy, R. C., Dutton, A. R., Kelley, V. A., Deeds, N. E. (2012). Sources of groundwater pumpage in a layered	Kelley, V. A., Deeds, N. E., Fryar, D. G., Nicol, J. P. (2004). Groundwater availability models for the Queen City and Sparta aquifers.	-	Texas portion approximated from Fig. 4 of Huang et al. (2012), with the eastern boundary approximated by the division

Aquifer System	Broader System	References			Delineation steps and sources
		aquifer system in the Upper Gulf Coastal Plain, USA. Hydrogeology Journal , 20(4), 783-796.	Contract report to the Texas Water Development Board, Austin, TX, 867 pp.		between Texas' 12th and 13th Groundwater Management Area (GMA; accessed February 11, 2021 from https://www.twdb.texas.gov/groundwater/management_areas/gma12.asp). The Mexico portion of the aquifer was approximated based on well depth data.
Tulare Basin	California Central Valley	Bertoldi, G. L., Johnston, R. H., Evenson, K. D. (1991). Ground water in the Central Valley, California: a summary report. United States Geological Survey Professional Paper 1401-A.	-	-	Approximated from Fig. 4 of Bertoldi et al. (1991); hydrologic regions (https://atlas-dwr.opendata.arcgis.com/datasets/2a572a181e094020bdaeb5203162de15_0) used to distinguish the three parts of the Central Valley
San Joaquin Basin	California Central Valley	Bertoldi, G. L., Johnston, R. H., Evenson, K. D. (1991). Ground water in the Central Valley, California: a summary report. United States Geological Survey Professional Paper 1401-A.	Mendenhall, W. C. (1908). Ground waters of the San Joaquin Valley, California. US Geological Survey Water Supply Paper 222, 54 pp. Accessed April 6, 2021 from https://pubs.usgs.gov/wsp/0222/report.pdf	-	Approximated from Fig. 4 of Bertoldi et al. (1991); hydrologic regions (https://atlas-dwr.opendata.arcgis.com/datasets/2a572a181e094020bdaeb5203162de15_0) used to distinguish the three parts of the Central Valley. Delineation of conditions also available via plate 1 by Mendenhall (1908).
Sacramento Basin	California Central Valley	Bertoldi, G. L., Johnston, R. H., Evenson, K. D. (1991). Ground water in the Central Valley, California: a summary report. United States Geological Survey Professional Paper 1401-A.	Mendenhall, W. C. (1908). Ground waters of the San Joaquin Valley, California. US Geological Survey Water Supply Paper 222, 54 pp. Accessed April 6, 2021 from https://pubs.usgs.gov/wsp/0222/report.pdf	-	Approximated from Fig. 4 of Bertoldi et al. (1991); hydrologic regions (https://atlas-dwr.opendata.arcgis.com/datasets/2a572a181e094020bdaeb5203162de15_0) used to distinguish the three parts of the Central Valley. Delineation of conditions also available via plate 1 by Mendenhall (1908).
Denver Basin	-	Ruybal, C. J., Hogue, T. S., McCray, J. E. (2019). Assessment of groundwater depletion and implications for management in the Denver Basin Aquifer System. JAWRA Journal of the American Water Resources Association , 55, 1130-1148.	Paschke, S.S. (2011). Groundwater availability of the Denver Basin aquifer system, Colorado. U.S. Geological Survey Professional Paper 1770, 274 pp. Accessed January 21, 2022 from https://pubs.usgs.gov/pp/1770/contents/pp1770.pdf	Malenda, H.F., Penn, C.A., (2020). Groundwater levels in the Denver Basin bedrock aquifers of Douglas County, Colorado, 2011–19. U.S. Geological Survey Scientific Investigations Report 2020–5076, 44 pp., Accessed November 29, 2021 from https://pubs.usgs.gov/sir/2020/5076/sir20205076.pdf	Approximated from Fig. A4 of Paschke et al. (2011), Fig. 1 by Malenda and Penn (2020), and Fig. 4 by Ruybal et al. (2019)

Aquifer System	Broader System	References	Delineation steps and sources		
West Salt River Basin	Salt Basin	Flora, S., Davis, T. (2009). Hydrologic Map Series (HMS), Water Level Change Map Series (WLCMS), and Basin Sweep Assessment Report ADWR Basins and Sub-Basins. Arizona Department of Water Resources Hydrology Division Field Services Section https://new.azwater.gov/sites/default/files/HMSWLCMSBasinSweepAssessmentReport2009.pdf	Anning, D. (2014). Conceptual Understanding and Groundwater Quality of Selected Basin-Fill Aquifers in the Southwestern United States: Section 7. Conceptual Understanding and Groundwater Quality of the Basin-Fill Aquifer in the West Salt River Valley, Arizona. U.S. Geological Survey Professional Paper 1781, 24 pp. Accessed March 31, 2021 from https://pubs.usgs.gov/pp/1781/pdf/pp1781_section7.pdf	-	Approximated from Fig. 1 of Flora and Davis (2009). Hydrologic Map Series (HMS), Water Level Change Map Series (WLCMS), and Basin Sweep Assessment Report ADWR Basins and Sub-Basins. Arizona Department of Water Resources Hydrology Division Field Services Section https://new.azwater.gov/sites/default/files/HMSWLCMSBasinSweepAssessmentReport2009.pdf . See also: Anning (2014).
East Salt River Basin	Salt Basin	Flora, S., Davis, T. (2009). Hydrologic Map Series (HMS), Water Level Change Map Series (WLCMS), and Basin Sweep Assessment Report ADWR Basins and Sub-Basins. Arizona Department of Water Resources Hydrology Division Field Services Section https://new.azwater.gov/sites/default/files/HMSWLCMSBasinSweepAssessmentReport2009.pdf	Laney, R.L., Hahn, M.E. (1986). Hydrogeology of the eastern part of the Salt River Valley area, Maricopa and Pinal Counties, Arizona. U.S. Geological Survey Water Resources Investigations Report Water Resources Investigations Report-8 maps on 4 sheets. Accessed March 31, 2021 from https://pubs.er.usgs.gov/publication/wri864447	-	Approximated from Fig. 1 of Flora and Davis (2009). Hydrologic Map Series (HMS), Water Level Change Map Series (WLCMS), and Basin Sweep Assessment Report ADWR Basins and Sub-Basins. Arizona Department of Water Resources Hydrology Division Field Services Section https://new.azwater.gov/sites/default/files/HMSWLCMSBasinSweepAssessmentReport2009.pdf . See also: Laney and Hahn (1986).
Maricopa-Stanfield Basin	-	Flora, S., Davis, T. (2009). Hydrologic Map Series (HMS), Water Level Change Map Series (WLCMS), and Basin Sweep Assessment Report ADWR Basins and Sub-Basins. Arizona Department of Water Resources Hydrology Division Field Services Section https://new.azwater.gov/sites/default/files/HMSWLCMSBasinSweepAssessmentReport2009.pdf	-	-	Approximated from Fig. 1 of Flora and Davis (2009). Hydrologic Map Series (HMS), Water Level Change Map Series (WLCMS), and Basin Sweep Assessment Report ADWR Basins and Sub-Basins. Arizona Department of Water Resources Hydrology Division Field Services Section https://new.azwater.gov/sites/default/files/HMSWLCMSBasinSweepAssessmentReport2009.pdf
Avra Valley	Tucson Basin and Avra Valley	Flora, S., Davis, T. (2009). Hydrologic Map Series (HMS), Water Level Change Map Series (WLCMS), and Basin Sweep Assessment Report ADWR Basins and Sub-Basins. Arizona Department of Water Resources Hydrology Division Field Services Section https://new.azwater.gov/sites/default/files/HMSWLCMSBasinSweepAssessmentReport2009.pdf	Carruth, R.L., Kahler, L.M., Conway, B.D. (2018). Groundwater storage change and land surface elevation change in Tucson Basin and Avra Valley, south-central Arizona—2003–2016. U.S. Geological Survey Scientific Investigations Report 2018–5154, 34 pp. Accessed April 1, 2021 from	Hanson, R.T. (1989). Aquifer system compaction, Tucson Basin and Avra Valley, Arizona. Water Resources Investigations Report 88-4172, 75 pp. Accessed April 15, 2021 from https://pubs.usgs.gov/wri/1988/4172/report.pdf	Approximated from Fig. 1 of Flora and Davis (2009). See also Fig. 1 of Carruth et al. (2018). Broader Tucson Basin and Avra Valley outline from Fig. 1 of Hanson (1989).

Aquifer System	Broader System	References	Delineation steps and sources
			https://pubs.usgs.gov/sir/2018/5154/sir20185154.pdf
Upper Santa Cruz Basin	Tucson Basin and Avra Valley	Flora, S., Davis, T. (2009). Hydrologic Map Series (HMS), Water Level Change Map Series (WLCMS), and Basin Sweep Assessment Report ADWR Basins and Sub-Basins. Arizona Department of Water Resources Hydrology Division Field Services Section https://new.azwater.gov/sites/default/files/HMSWLCMSBasinSweepAssessmentReport2009.pdf	Goes, A., Gellenbeck, D.J., Towne, D.C., Freark, M.C. (2002). Ground water quality in the Upper Santa Cruz Basin, U.S. Geological Survey Water Resources Investigations Report 00-4117, 66 pp. Accessed March 31, 2021 from https://pubs.usgs.gov/wri/2000/4117/report.pdf Hanson, R.T. (1989). Aquifer-system compaction, Tucson Basin and Avra Valley, Arizona. Water Resources Investigations Report 88-4172, 75 pp. Accessed April 15, 2021 from https://pubs.usgs.gov/wri/1988/4172/report.pdf Approximated from Fig. 4 of Flora and Davis (2009). See also Coes et al. (2002). Broader Tucson Basin and Avra Valley outline from Fig. 4 of Hanson (1989).
Transboundary Santa Cruz Basin	-	Callegary, J.B., Minjárez Sosa, I., Tapia Villaseñor, E.M., dos Santos, P., Monreal Saavedra, R., Grijalva Noriega, F.J., Huth, A.K., Gray, F., Scott, C.A., Megdal, S.B., Oroz Ramos, L.A., Rangel Medina, M., Leenhouts, J.M. (2016) San Pedro River Aquifer Binational Report: International Boundary and Water Commission. 173 pp. Accessed February 12, 2021 via https://www.ibwc.gov/Files/Binational_Study_Transboundary_San_Pedro_Aquifer.pdf	Sanchez, R., Lopez, V., Eckstein, G. (2016). Identifying and characterizing transboundary aquifers along the Mexico-US border: An initial assessment. Journal of Hydrology, 535, 101-119. Flora, S., Davis, T. (2009). Hydrologic Map Series (HMS), Water Level Change Map Series (WLCMS), and Basin Sweep Assessment Report ADWR Basins and Sub-Basins. Arizona Department of Water Resources Hydrology Division Field Services Section https://new.azwater.gov/sites/default/files/HMSWLCMSBasinSweepAssessmentReport2009.pdf . Mexico Portion estimate from Fig. 3 of Sanchez et al., 2016
San Pedro Basin	-	Callegary, J.B., Minjárez Sosa, I., Tapia Villaseñor, E.M., dos Santos, P., Monreal Saavedra, R., Grijalva Noriega, F.J., Huth, A.K., Gray, F., Scott, C.A., Megdal, S.B., Oroz Ramos, L.A., Rangel Medina, M., Leenhouts, J.M. (2016) San Pedro River Aquifer Binational Report: International Boundary and Water Commission. 173 pp. Accessed February 12, 2021 via https://www.ibwc.gov/Files/Binational_Study_Transboundary_San_Pedro_Aquifer.pdf	Sanchez, R., Lopez, V., Eckstein, G. (2016). Identifying and characterizing transboundary aquifers along the Mexico-US border: An initial assessment. Journal of Hydrology, 535, 101-119. Flora, S., Davis, T. (2009). Hydrologic Map Series (HMS), Water Level Change Map Series (WLCMS), and Basin Sweep Assessment Report ADWR Basins and Sub-Basins. Arizona Department of Water Resources Hydrology Division Field Services Section https://new.azwater.gov/sites/default/files/HMSWLCMSBasinSweepAssessmentReport2009.pdf . Mexico Portion estimate from Fig. 3 of Sanchez et al. (2016)
Ciénega Basin	-	Flora, S., Davis, T. (2009). Hydrologic Map Series (HMS), Water Level Change Map Series (WLCMS), and Basin Sweep Assessment Report ADWR Basins and	- Ciénega Basin approximated from Fig. 4 of Flora and Davis (2009). Hydrologic Map Series (HMS), Water Level Change Map Series

Aquifer System	Broader System	References	Delineation steps and sources
		Sub-Basins. Arizona Department of Water Resources Hydrology Division Field Services Section https://new.azwater.gov/sites/default/files/HMSWLCMSBasinSweepAssessmentReport2009.pdf	(WLCMS), and Basin Sweep Assessment Report ADWR Basins and Sub-Basins. Arizona Department of Water Resources Hydrology Division Field Services Section https://new.azwater.gov/sites/default/files/HMSWLCMSBasinSweepAssessmentReport2009.pdf
Willcox-Douglas Basin	-	Arizona Department of Water Resources (2018). Groundwater Flow Model of the Willcox Basin, Arizona Department of Water Resources Report, 196 pp. Accessed on February 12, 2021 from https://new.azwater.gov/sites/default/files/Wilcox_Report_2018.pdf	Flora, S., Davis, T. (2009). Hydrologic Map Series (HMS), Water Level Change Map Series (WLCMS), and Basin Sweep Assessment Report ADWR Basins and Sub-Basins. Arizona Department of Water Resources Hydrology Division Field Services Section https://new.azwater.gov/sites/default/files/HMSWLCMSBasinSweepAssessmentReport2009.pdf
Mammoth Basin	-	Flora, S., Davis, T. (2009). Hydrologic Map Series (HMS), Water Level Change Map Series (WLCMS), and Basin Sweep Assessment Report ADWR Basins and Sub-Basins. Arizona Department of Water Resources Hydrology Division Field Services Section https://new.azwater.gov/sites/default/files/HMSWLCMSBasinSweepAssessmentReport2009.pdf	Approximated from Arizona Department of Water Resources (2018). This report states (quote AZ DWR): "...the Willcox Basin is a closed basin that is in direct hydraulic connection with the Aravaipa and Douglas basins to the north and south, respectively." And thus the Aravaipa and Douglas Basins of Flora and Davis (2009) were included in the delineated boundary for the Willcox Basin.
Yuma Basin	-	Olmsted, F. H., Loeltz, O. J., Irelan, B. (1973). Geohydrology of the Yuma area, Arizona and California. US Geological Survey Professional Paper 486-H, 237 pp. Accessed February 15, 2021 from https://pubs.usgs.gov/pp/0486h/report.pdf	Approximated from Fig. 2 of Olmsted et al. (1973); southwestern margin approximated by Colorado River
Ozark Plateaus Aquifer System	-	Clark, B. R., Duncan, L. L., Knierim, K. J. (2016). Groundwater availability in the Ozark Plateaus aquifer system (No. 1854). US Geological Survey Professional Paper 1854, 95 pp. Accessed February 13, 2021 from https://pubs.er.usgs.gov/publication/pp1854	Imes, J. L., Emmett, L. F. (1994). Geohydrology of the Ozark Plateaus aquifer system in parts of Missouri, Arkansas, Oklahoma, and Kansas (No. 1414-D). Accessed February 13, 2021 from https://pubs.usgs.gov/pp/1414d/report.pdf
			Hays, P. D., Knierim, K. J., Breaker, Brian, Westerman, D. A., and Clark, B. R., 2016, Hydrogeology and hydrologic conditions of the Ozark Plateaus aquifer system: U.S. Geological Survey Scientific Investigations Report 2016-5137, 61 p.,

Aquifer System	Broader System	References		Delineation steps and sources	
				http://dx.doi.org/10.3133/sir20166137 .	
Los Angeles Basin	-	Land, M., Reichard, E.G., Crawford, S.M., Everett, R.R., Newhouse, M.W., Williams, C.F. (2004). Ground-water Quality of Coastal Aquifer Systems in the West Coast Basin, Los Angeles County, California, 1999–2002. U.S. Geological Survey Scientific Investigations Report 2004-5067, 88 pp. Accessed February 15, 2021 from https://pubs.usgs.gov/sir/2004/5067/sir2004-5067.pdf	Fram, M. S., Belitz, K. (2008). Groundwater quality in the Coastal Los Angeles Basin, California; United States Geological Survey Fact Sheet 2012-3096. Accessed February 14, 2021 from https://pubs.er.usgs.gov/publication/70039952 .	Reichard, E. G., Land, M., Crawford, S.M., Johnson, T.D., Everett, R.R., Kulshan, T.V., Ponti, D.J., Halford, K.L., Johnson, T.A., Paybine, K.S., Nishikawa, T. (2003). Geohydrology, Geochemistry, and Ground-Water Simulation Optimization of the Central and West Coast Basins, Los Angeles County, California. USGS Numbered Series, 2003-4065. Accessed November 28, 2021 from https://pubs.usgs.gov/wri/wri034065/wri034065.pdf	Approximated from Fig. 1 by Land et al. (2004), Fig. 6 by Reichard et al. (2003) and map on page 1 of Fram and Belitz (2008)
Salinas Valley	-	Salinas Valley Basin Integrated Sustainability Plan (2020). Accessed February 15, 2021 from https://evbgsa.org/wp-content/uploads/2019/03/Valley-Wide-Integrated-Sustainability-Plan-optimized.pdf	Hamlin, H. (1904). Water resources of the Salinas Valley, California. U.S. Geological Survey Water Supply Paper 89, 123 pp. Accessed April 6, 2021 from https://pubs.usgs.gov/wsp/0089/report.pdf	-	Approximated from Fig. 5-2 of Salinas Valley Basin Integrated Sustainability Plan (2020) and Plate 1 of Hamlin (1904).
Coachella Valley	-	Sneed, M., Brandt, J.T., Solt, M. (2014). Land subsidence, groundwater levels, and geology in the Coachella Valley, California, 1993–2010. U.S. Geological Survey, Scientific Investigations Report 2014-5075, 62 pp. Accessed February 15, 2021 from https://pubs.usgs.gov/sir/2014/5075/pdf/sir2014-5075.pdf	-	-	Approximated from Figs. 1 and 2 of Sneed et al. (2014)
Williston Basin	Northern Great Plains	Thamke, J. N., LeCain, G. D., Ryter, D. W., Sando, R., Long, A. J. (2014). Hydrogeologic framework of the uppermost principal aquifer systems in the Williston and Powder River structural basins, United States and Canada. US Geological Survey Scientific Investigations Report 2014-5047, 50 pp. Accessed February 15, 2021 from https://pubs.usgs.gov/sir/2014/5047/pdf/sir2014-5047.pdf	Long, A.J., Thamke, J.N., Davis, K.W., Bartos, T.T. (2018). Groundwater availability of the Williston Basin, United States and Canada. U.S. Geological Survey Professional Paper 1841, 42 p. Accessed February 16, 2021 from https://doi.org/10.3133/pp1841	-	Approximated from Fig. 1 of Long et al. (2018) and Fig. 1 by Thamke et al. (2014); "Miles City arch" boundary used to distinguish Williston and Powder River structural basins
Powder River Basin	Northern Great Plains	Thamke, J. N., LeCain, G. D., Ryter, D. W., Sando, R., Long, A. J. (2014). Hydrogeologic framework of the	Long, A.J., Thamke, J.N., Davis, K.W., Bartos, T.T. (2018). Groundwater availability of the	-	Approximated from Fig. 1 of Long et al. (2018) and Fig. 1 by Thamke et al. (2014); "Miles City arch"

Aquifer System	Broader System	References	Delineation steps and sources		
		uppermost principal aquifer systems in the Williston and Powder River structural basins, United States and Canada. US Geological Survey Scientific Investigations Report 2014-5047, 50 pp. Accessed February 16, 2021 from https://pubs.usgs.gov/sir/2014/5047/pdf/i2014-5047.pdf	Williston Basin, United States and Canada: U.S. Geological Survey Professional Paper 1841, 42 p., Accessed February 16, 2021 from https://doi.org/10.3133/pp1841	boundary used to distinguish Williston and Powder River structural basins	
Black Hills Uplift	-	Driscoll, D. G., Carter, J. M., Williamson, J. E., Putnam, L. D. (2002). Hydrology of the Black Hills area, South Dakota. US Geological Survey Water Resources Investigations Report 2002-4094, 158 pp. Accessed February 16, 2021 from https://pubs.usgs.gov/wri/wri024094/pdf/wri024094.pdf	-	Approximated from Fig. 2 of Driscoll et al. (2002)	
Southern Willamette Valley	Willamette Valley	Woodward, D. G., Gannett, M. W., Vaccaro, J. J. (1998). Hydrogeologic framework of the Willamette Lowland aquifer system, Oregon and Washington. US Geological Survey Professional Paper 1424-B, 92 pp. Accessed February 16, 2021 from https://pubs.usgs.gov/pp/1424b/report.pdf	Herrera, N. B., Burns, E. R., Conlon, T. D. (2014). Simulation of groundwater flow and the interaction of groundwater and surface water in the Willamette Basin and Central Willamette Subbasin, Oregon. US Geological Survey Scientific Investigations Report 2014-5136, 174 pp. Accessed February 16, 2021 from https://pubs.usgs.gov/sir/2014/5136/pdf/sir20145136.pdf	Approximated from Fig. 2 of Woodward et al. (1998). Basin extent was broadened to include foothills of surrounding mountains, as a considerable number of wells exist in these areas. The Waldo and Salem Hills were used to divide this southern portion of the aquifer system from the Central Willamette Valley to the north.	
Central Willamette Valley	Willamette Valley	Woodward, D. G., Gannett, M. W., Vaccaro, J. J. (1998). Hydrogeologic framework of the Willamette Lowland aquifer system, Oregon and Washington. US Geological Survey Professional Paper 1424-B, 92 pp. Accessed February 16, 2021 from https://pubs.usgs.gov/pp/1424b/report.pdf	Herrera, N. B., Burns, E. R., Conlon, T. D. (2014). Simulation of groundwater flow and the interaction of groundwater and surface water in the Willamette Basin and Central Willamette Subbasin, Oregon. US Geological Survey Scientific Investigations Report 2014-5136, 174 pp. Accessed February 16, 2021 from https://pubs.usgs.gov/sir/2014/5136/pdf/sir20145136.pdf	Approximated from Fig. 2 of Woodward et al. (1998). Basin extent was broadened to include foothills of surrounding mountains, as a considerable number of wells exist in these areas. The Waldo and Salem Hills were used to divide this Central Willamette Valley from the Southern Willamette Valley to the south. The Chehalem Mountains were used as an approximate divide between this central portion of the aquifer system and the Tualatin Basin to the north.	
Tualatin and Portland Basins	Willamette Valley	Woodward, D. G., Gannett, M. W., Vaccaro, J. J. (1998). Hydrogeologic framework of the Willamette Lowland aquifer system, Oregon and Washington. US Geological Survey Professional	Herrera, N. B., Burns, E. R., Conlon, T. D. (2014). Simulation of groundwater flow and the interaction of groundwater and surface water in the Willamette Basin and Central	Jones, M. A. (1999). Geologic framework for the Puget Sound aquifer system, Washington and British Columbia. US Geological	Approximated from Fig. 2 of Woodward et al. (1998). Basin extent was broadened to include foothills of surrounding mountains, as a considerable number of wells

Aquifer System	Broader System	References	Delineation steps and sources		
		Paper 1424-B, 92 pp. Accessed February 16, 2021 from https://pubs.usgs.gov/pp/1424b/report.pdf	Willamette Subbasin, Oregon, US Geological Survey Scientific Investigations Report 2014-5136, 174 pp. Accessed February 16, 2021 from https://pubs.usgs.gov/sir/2014/5136/pdf/sir20145136.pdf	Survey Professional Report 1424-C, 44 pp. Accessed February 25, 2021 from https://pubs.usgs.gov/pp/1424c/report.pdf	exist in these areas. The Chehalem Mountains were used as an approximate divide between this Tualatin Basin and the Central Willamette Valley to the south. The northern extent of the Basin crosses the Columbia River and is based on the approximate northern limit delineated in Fig. 1 of Jones (1999) (see page 12 or "C4" of report).
Palouse Basin	Columbia Plateau Regional Aquifer System	Douglas, A. A., Osiensky, J. L., Keller, C. K. (2007). Carbon-14 dating of ground water in the Palouse Basin of the Columbia River basalts. Journal of Hydrology, 334(3-4), 502-512.	Kahle, S.C., Morgan, D.S., Welch, W.B., Ely, S.R., Vaccaro, J.J., Orzol, L.L. (2011). Hydrogeologic Framework and Hydrologic Budget Components of the Columbia Plateau Regional Aquifer System, Washington, Oregon, and Idaho. US Geological Survey Scientific Investigations Report 2011-5124, 80 pp. Accessed February 16, 2021 from https://pubs.usgs.gov/sir/2011/5124/pdf/sir20115124.pdf	Reidel, S. P., Spane, F. A., Johnson, V. G. (2002). Natural gas storage in basalt aquifers of the Columbia basin, Pacific Northwest USA: A guide to site characterization. Pacific Northwest National Lab (PNNL) Report n.No. PNNL-13962, Richland, Washington, USA, 277 pp. Accessed February 16, 2021 from https://www.pnnl.gov/main/publications/external/technical_reports/PNNL-13962.pdf	Palouse Basin approximated based on Fig. 1 of Douglas et al. (2007). Broader Columbia River Aquifer System approximated from Figs. 1 and 10 of Kahle et al. (2011) and sub-province map shown in Fig. 2.1 of Reidel et al. (2002).
Yakima Basin	Columbia Plateau Regional Aquifer System	Burns, E. R., Snyder, D. T., Haynes, J. V., Waibel, M. S. (2012). Groundwater status and trends for the Columbia Plateau Regional Aquifer System, Washington, Oregon, and Idaho. U.S. Geological Survey Scientific Investigations Report 2012-5261, 52 pp. Accessed February 18, 2021 from http://pubs.usgs.gov/publication/sir20125261/.	Kahle, S.C., Morgan, D.S., Welch, W.B., Ely, S.R., Vaccaro, J.J., Orzol, L.L. (2011). Hydrogeologic Framework and Hydrologic Budget Components of the Columbia Plateau Regional Aquifer System, Washington, Oregon, and Idaho. US Geological Survey Scientific Investigations Report 2011-5124, 80 pp. Accessed February 16, 2021 from https://pubs.usgs.gov/sir/2011/5124/pdf/sir20115124.pdf	Reidel, S. P., Spane, F. A., Johnson, V. G. (2002). Natural gas storage in basalt aquifers of the Columbia basin, Pacific Northwest USA: A guide to site characterization. Pacific Northwest National Lab (PNNL) Report n.No. PNNL-13962, Richland, Washington, USA, 277 pp. Accessed February 16, 2021 from https://www.pnnl.gov/main/publications/external/technical_reports/PNNL-13962.pdf	Yakima Basin approximated based on Fig. 7 of Burns et al. (2012). Broader Columbia River Aquifer System approximated from Figs. 1 and 10 of Kahle et al. (2011) and sub-province map shown in Fig. 2.1 of Reidel et al. (2002).
Odessa Subregion	Columbia Plateau Regional Aquifer System	United States Bureau of Reclamation (2012). Final Feasibility Level Special Study Report Odessa Subarea Special Study. 244 pp. Accessed February 18, 2021 from	Kahle, S.C., Morgan, D.S., Welch, W.B., Ely, S.R., Vaccaro, J.J., Orzol, L.L. (2011). Hydrogeologic Framework and Hydrologic Budget Components of the Columbia	Vaccaro, J.J., Kahle, S.C., Ely, D.M., Burns, E.R., Snyder, D.T., Haynes, J.V., Olsen, T.D., Welch, W.B., and Morgan, D.S., 2015,	Odessa subarea approximated from location map on page 75 (Fig. 4-10) of US Bureau of Reclamation (2012) and Fig. 38 of Vaccaro et al. (2015). Broader Columbia River

Aquifer System	Broader System	References	Delineation steps and sources		
		https://www.usbr.gov/pn/programs/eis/odessa/fina/eis/final.pdf	Plateau Regional Aquifer System, Washington, Oregon, and Idaho-US Geological Survey Scientific Investigations Report 2011-5124, 80 pp. Accessed February 16, 2021 from https://pubs.usgs.gov/sir/2011/5124/pdf/sir20115124.pdf	Groundwater availability of the Columbia Plateau Regional Aquifer System, Washington, Oregon, and Idaho-U.S. Geological Survey Professional Paper 1847, 87 pp. Accessed February 18, 2021 from http://dx.doi.org/10.3133/pp1847.	Aquifer System approximated from Figs. 1 and 40 of Kahle et al. (2014).
Quincy Subregion	Columbia Plateau Regional Aquifer System	United States Bureau of Reclamation (2012). Final Feasibility Level Special Study Report Odessa Subarea Special Study. 244 pp. Accessed February 18, 2021 from https://www.usbr.gov/pn/programs/eis/odessa/fina/eis/final.pdf	Kahle, S.C., Morgan, D.S., Welch, W.B., Ely, S.R., Vaccaro, J.J., Orzol, L.L. (2011). Hydrogeologic Framework and Hydrologic Budget Components of the Columbia Plateau Regional Aquifer System, Washington, Oregon, and Idaho-US Geological Survey Scientific Investigations Report 2011-5124, 80 pp. Accessed February 16, 2021 from https://pubs.usgs.gov/sir/2011/5124/pdf/sir20115124.pdf	Vaccaro, J.J., Kahle, S.C., Ely, D.M., Burns, E.R., Snyder, D.T., Haynes, J.V., Olsen, T.D., Welch, W.B., and Morgan, D.S. (2015). Groundwater availability of the Columbia Plateau Regional Aquifer System, Washington, Oregon, and Idaho-U.S. Geological Survey Professional Paper 1847, 87 pp. Accessed February 18, 2021 from http://dx.doi.org/10.3133/pp1847.	Odessa subarea approximated from location map on page 75 (Fig. 4-10) of US Bureau of Reclamation (2012) and Fig. 38 of Vaccaro et al. (2015). Broader Columbia River Aquifer System approximated from Figs. 1 and 40 of Kahle et al. (2014).
Northern Columbia River Basin	Columbia Plateau Regional Aquifer System	Kahle, S.C., Morgan, D.S., Welch, W.B., Ely, S.R., Vaccaro, J.J., Orzol, L.L. (2011). Hydrogeologic Framework and Hydrologic Budget Components of the Columbia Plateau Regional Aquifer System, Washington, Oregon, and Idaho-US Geological Survey Scientific Investigations Report 2011-5124, 80 pp. Accessed February 16, 2021 from https://pubs.usgs.gov/sir/2011/5124/pdf/sir20115124.pdf	-	-	Columbia River Aquifer System approximated from Figs. 1 and 40 of Kahle et al. (2014).
Spokane Valley-Rathdrum Prairie Aquifer	Columbia Plateau Regional Aquifer System	Kahle, S.C., Morgan, D.S., Welch, W.B., Ely, S.R., Vaccaro, J.J., Orzol, L.L. (2014). Hydrogeologic Framework and Hydrologic Budget Components of the Columbia Plateau Regional Aquifer System, Washington, Oregon, and Idaho-US Geological Survey Scientific Investigations Report 2011-5124, 80 pp. Accessed February 16, 2021 from https://pubs.usgs.gov/sir/2011/5124/pdf/sir20115124.pdf	Hsieh, P. A., Barber, M. E., Conter, B. A., Hossain, M., Johnson, G. S., Jones, J. L., Wylie, A. H. (2007). Ground-water flow model for the Spokane valley-Rathdrum prairie aquifer, Spokane County, Washington, and Bonner and Kootenai Counties, Idaho. U. S. Geological Survey Scientific Investigations Report 2007-5044, 90 pp. Accessed February 18, 2021 from	Graham, W. G., Campbell, L. J. (1984). Groundwater resources of Idaho. Idaho Department of Water Resources Report, 64 pp. Accessed March 23, 2021 from https://idwr.idaho.gov/files/publications/198408-MISC-GW-Resources-ID.pdf	Spokane Valley-Rathdrum Prairie Aquifer approximated from Fig. 4 of Hsieh et al. (2007). Columbia River Aquifer System approximated from Figs. 1 and 40 of Kahle et al. (2014). Idaho portion delineated from Graham and Campbell (1984).

Aquifer System	Broader System	References		Delineation steps and sources	
			https://pubs.usgs.gov/sir/2007/5044/pdf/sir20075044.pdf		
Little Spokane Basin	Columbia Plateau Regional Aquifer System	Kahle, S.C., Morgan, D.S., Welch, W.B., Ely, S.R., Vaccaro, J.J., Orzol, L.L. (2014). Hydrogeologic Framework and Hydrologic Budget Components of the Columbia Plateau Regional Aquifer System, Washington, Oregon, and Idaho. US Geological Survey Scientific Investigations Report 2011-5124, 80 pp. Accessed February 16, 2021 from https://pubs.usgs.gov/sir/2011/5124/pdf/sir20115124.pdf	Kahle, S. C., Olsen, T. D., Fasser, E. T. (2013). Hydrogeology of the Little Spokane River Basin, Spokane, Stevens, and Pend Oreille Counties, Washington. US Geological Survey Scientific Investigations Report 2013-5124, 64 pp. Accessed February 18, 2021 from https://pubs.usgs.gov/sir/2013/5124/pdf/sir20135124.pdf	-	Little Spokane Basin approximated from Fig. 5 of Kahle et al. (2012). Columbia River Aquifer System approximated from Figs. 1 and 40 of Kahle et al. (2011).
Blue Mountains and Clearwater Embayment	Columbia Plateau Regional Aquifer System	Kahle, S.C., Morgan, D.S., Welch, W.B., Ely, S.R., Vaccaro, J.J., Orzol, L.L. (2014). Hydrogeologic Framework and Hydrologic Budget Components of the Columbia Plateau Regional Aquifer System, Washington, Oregon, and Idaho. US Geological Survey Scientific Investigations Report 2011-5124, 80 pp. Accessed February 16, 2021 from https://pubs.usgs.gov/sir/2011/5124/pdf/sir20115124.pdf	Graham, W. G., Campbell, L. J. (1981). Groundwater resources of Idaho. Idaho Department of Water Resources Report, 61 pp. Accessed March 23, 2021 from https://idwr.idaho.gov/files/publications/198108-MISC-GW-Resources-ID.pdf	-	Physiographic Provinces (Blue Mountains, Clearwater Embayment) approximated from Fig. 1 of Kahle et al. (2011). Idaho portion approximated following Plate 1 of Graham and Campbell (1981; see "Clearwater Uplands (Mussel Shell Basalt Subsection)"; Columbia River Aquifer System approximated from Figs. 1 and 40 of Kahle et al. (2011).
Palouse Slope	Columbia Plateau Regional Aquifer System	Kahle, S.C., Morgan, D.S., Welch, W.B., Ely, S.R., Vaccaro, J.J., Orzol, L.L. (2014). Hydrogeologic Framework and Hydrologic Budget Components of the Columbia Plateau Regional Aquifer System, Washington, Oregon, and Idaho. US Geological Survey Scientific Investigations Report 2011-5124, 80 pp. Accessed February 16, 2021 from https://pubs.usgs.gov/sir/2011/5124/pdf/sir20115124.pdf	-	-	Physiographic Provinces (Blue Mountains, Clearwater Embayment) approximated from Fig. 1 of Kahle et al. (2011). Columbia River Aquifer System approximated from Figs. 1 and 40 of Kahle et al. (2011).
Walla Walla Basin	Columbia Plateau Regional Aquifer System	Kahle, S.C., Morgan, D.S., Welch, W.B., Ely, S.R., Vaccaro, J.J., Orzol, L.L. (2014). Hydrogeologic Framework and Hydrologic Budget Components of the Columbia Plateau Regional Aquifer System, Washington, Oregon, and Idaho. US Geological Survey Scientific Investigations Report 2011-5124, 80 pp. Accessed February 16, 2021 from https://pubs.usgs.gov/sir/2011/5124/pdf/sir20115124.pdf	Henry, R., Lindsay, K., Wolcott, B., Patten, S., Baker, T. (2013). Walla Walla Basin Aquifer Recharge Strategic Plan. Walla Walla Basin Watershed Council Report, 106 pp. Accessed February 23, 2021 from http://www.wbc.org/images/Projects/AR/Reports/RechargeStrategy_FINAL_1-29-13_sp.pdf	Newcomb, R.C. (1965). Geology and ground-water resources of the Walla Walla River Basin, Washington. Oregon-Washington Division of Water Resources Water Supply Bulletin 21, 162 pp. Accessed November 29, 2021 from https://apps.ecology.wa.gov/publications/documents/wsb21.pdf	Walla Walla Basin outline approximated from Fig. 1 of Walla Walla Basin Watershed Council Report by Henry et al. (2013) and Fig. 1 by Newcomb (1965). Columbia River Aquifer System approximated from Figs. 1 and 40 of Kahle et al. (2011).

Aquifer System	Broader System	References	Delineation steps and sources		
Umatilla Basin and Horse Heaven Hills	Columbia Plateau Regional Aquifer System	Kahle, S.C., Morgan, D.S., Welch, W.B., Ely, S.R., Vaccaro, J.J., Orzol, L.L. (2011). Hydrogeologic Framework and Hydrologic Budget Components of the Columbia Plateau Regional Aquifer System, Washington, Oregon, and Idaho. US Geological Survey Scientific Investigations Report 2011-5124, 80 pp. Accessed February 16, 2021 from https://pubs.usgs.gov/sir/2011/5124/pdf/sir20115124.pdf	Herrera, N. B., Ely, K., Mehta, S., Stonewall, A. J., Risley, J. C., Hinkle, S. R., Conlon, T. D. (2017). Hydrogeologic framework and selected components of the groundwater budget for the upper Umatilla River Basin, Oregon (No. 2017-5020). US Geological Survey Scientific Investigations Report 2017-5020, 68 pp. Accessed February 18, 2021 from https://pubs.usgs.gov/sir/2017/5020/sir20175020.pdf	Davies-Smith, A., Bolke, E. L., Collins, C. A. (1988). Geohydrology and digital simulation of the ground-water flow system in the Umatilla Plateau and Horse Heaven Hills area, Oregon and Washington. Water Resources Investigations Report 87-4268, 77 pp. Accessed February 23, 2021 from https://pubs.usgs.gov/wri/1987/4268/report.pdf	Approximate Umatilla Basin boundaries from Fig. 9 of Kahle et al. (2011); see also Fig. 3 of Herrera et al. (2017). Horse Heaven Hills extent (and co-study with Umatilla Basin) from Fig. 1 of Davies-Smith et al. (1983). Columbia River Aquifer System approximated from Figs. 1 and 10 of Kahle et al. (2011).
Klickitat Valley	Columbia Plateau Regional Aquifer System	Kahle, S.C., Morgan, D.S., Welch, W.B., Ely, S.R., Vaccaro, J.J., Orzol, L.L. (2011). Hydrogeologic Framework and Hydrologic Budget Components of the Columbia Plateau Regional Aquifer System, Washington, Oregon, and Idaho. US Geological Survey Scientific Investigations Report 2011-5124, 80 pp. Accessed February 16, 2021 from https://pubs.usgs.gov/sir/2011/5124/pdf/sir20115124.pdf	Luzier, J.E. (1969). Ground-water occurrence in the Goldendale area, Klickitat County, Washington. US Geological Survey Numbered Series, Hydrologic Atlas. Accessed February 23, 2021 from https://pubs.er.usgs.gov/publication/ha313	-	Approximate Klickitat Valley boundary from Plate 1 of Luzier (1969). Columbia River Aquifer System approximated from Figs. 1 and 10 of Kahle et al. (2011).
Lower Deschutes Area	Columbia Plateau Regional Aquifer System	Deschutes River Conservancy (2021). Webpage entitled "The Deschutes River Basin" accessed October 9, 2021 from https://www.deschutesriver.org/where-we-work/	-	-	Approximated from Deschutes River Conservancy (2021).
Albuquerque Basin	Middle Rio Grande	Bexfield, L. M., Anderholm, S. K. (2000). Predevelopment water level map of the Santa Fe Group aquifer system in the middle Rio Grande basin between Cochiti Lake and San Acacia, New Mexico. U.S. Geological Survey Water Resources Investigations Report 00-4249, 1 sheet. Accessed February 17, 2021 from https://doi.org/10.3133/wri004249 .	Bartolino, J.R., Cole, J.C. (2002). Ground-water resources of the Middle Rio Grande Basin. U.S. Geological Survey Water Resources Circular 1222, 145 pp. Accessed November 28, 2021 from https://pubs.usgs.gov/circ/2002/circ1222/pdf/circ1222.pdf	Hawley, J. W., Haase, C.S., Lozinsky, R.P. (1995). An Underground View of the Albuquerque Basin. Report No. CONF-9411293-TRN: IM9704%%261, 37-55. Accessed November 28, 2021 from https://nmwri.nmou.edu/wp-content/uploads/2015/watcon/proc39/Hawley.pdf	Approximated from map by Bexfield and Anderholm (2000); Fig. 1 by Hawley et al. (1995) and Fig. 2.1 by Bartolino and Cole (2002)
Espanola Basin	Middle Rio Grande	Land, L. (2016). Overview of Fresh and Brackish Water Quality in New Mexico. Open file Report 583-4 pp. Accessed February 17, 2021 from https://geoinfo.nmt.edu/resources/water/amp/brochures/BWA/Estancia_Basin_FB_WQNM.pdf	Manning, A.H. (2009). Ground-water temperature, noble gas, and carbon isotope data from the Espanola Basin, New Mexico. U.S. Geological Survey Scientific Investigations Report 2008-5200, 78 pp. Accessed November 28, 2021 from	-	Approximated from map by Land (2016), Fig. 1 by Manning (2009)

Aquifer System	Broader System	References			Delineation steps and sources
			https://pubs.usgs.gov/sir/2008/5200/pdf/SIR08-5200.pdf		
San Luis Valley	Middle Rio Grande	Land, L. (2016). Overview of Fresh and Brackish Water Quality in New Mexico. Open file Report 583. 4 pp. Accessed February 17, 2024 from https://geoinfo.nmt.edu/resources/water/amp/brochures/BWA/Estancia_Basin_FB_WQNM.pdf	-	-	Approximated from Map by Land (2016)
Roswell Basin	-	Land, L. (2016). Overview of Fresh and Brackish Water Quality in New Mexico. Open file Report 583. 4 pp. Accessed February 17, 2024 from https://geoinfo.nmt.edu/resources/water/amp/brochures/BWA/Estancia_Basin_FB_WQNM.pdf	-	-	Approximated from Map by Land (2016)
Valle de Juarez and Hueco Bolson	Tularosa-Hueco	Sanchez, R., Eckstein, C. (2020). Groundwater management in the borderlands of Mexico and Texas: The beauty of the unknown, the negligence of the present, and the way forward. Water Resources Research , 56(3), e2019WR026068.	Day, J. C. (1978). International Aquifer Management: The Hueco Bolson on the Rio Grande River. Natural Resources Journal , 163-180.	-	Approximated from Fig. 1 of Day (1978) and Fig. 2 of Sanchez and Eckstein (2020).
Tularosa Basin	Tularosa-Hueco	McLean, J. S. (1970). Saline ground-water resources of the Tularosa basin, New Mexico (No. 561). US Geological Survey. Accessed February 17, 2024 from https://pubs.usgs.gov/unnumbered/7013/9928/report.pdf	Huff, G. F. (2005). Simulation of ground-water flow in the basin-fill aquifer of the Tularosa Basin, south-central New Mexico, predevelopment through 2040 (Vol. 4, No. 4). US Geological Survey Scientific Investigations Report 2004-5197, 108 pp. Accessed February 17, 2024 from https://pubs.usgs.gov/sir/2004/5197/pdf/sir20045197.pdf	-	Approximated from Fig. 1 of McLean (1970)
Mesilla Valley	Rincon-Mesilla Valleys	Nickerson, E. L., Myers, R. G. (1993). Geohydrology of the Mesilla ground-water basin, Dona Ana County, New Mexico, and El Paso County, Texas (No. 92-4156). US Geological Survey, Water Resources Division; US Geological Survey Water Resources Investigations Report 92-4156. Accessed February 17, 2024 from https://pubs.usgs.gov/wri/1992/4156/report.pdf	Hawley, J. W., Lozinsky, R. P. (1992). Hydrogeologic framework of the Mesilla Basin in New Mexico and western Texas. New Mexico Bureau of Mines and Mineral Resources Open File Report 323, 95 pp. Accessed November 29, 2021 from https://geoinfo.nmt.edu/publications/openfile/downloads/300-399/323/ofr_323.pdf https://geoinfo.nmt.edu/publications/openfile/downloads/300-399/323/ofr_323.pdf	-	Approximated from map on page 40 of Nickerson and Myers (1993) and Fig. 1 by Hawley and Lozinsky (1993)
Rincon Valley	Rincon-Mesilla Valleys	Fuchs, E. H., King, J. P., Carroll, K. C. (2019). Quantifying disconnection of	-	-	Approximated from Fig. 2 of Fuchs et al. (2019)

Aquifer System	Broader System	References	Delineation steps and sources
Estancia Basin	-	groundwater from managed ephemeral surface water during drought and conjunctive agricultural use. Water Resources Research , 55(7), 5871-5890. Land, L. (2016). Overview of Fresh and Brackish Water Quality in New Mexico. Open file Report 583-4 pp. Accessed February 17, 2021 from https://geoinfo.nmt.edu/resources/water/amp/brochures/BWA/Estancia_Basin_FB_WQNM.pdf	Approximated from Map by Land (2016)
Southern San Juan Basin	-	Kernodle, J. M. (1996). Hydrogeology and steady state simulation of groundwater flow in the San Juan Basin, New Mexico, Colorado, Arizona, and Utah. US Geological Survey Water Resources Investigations Report 95-4187, 126 pp. Accessed March 12, 2021 from https://pubs.usgs.gov/wri/1995/4187/report.pdf	Approximated from Fig. 3 of Kernodle (1996). Southern portion approximated on the basis of groundwater well density (derived from New Mexico's well completion dataset).
Upper Deschutes Basin	-	Gannett, M.W., Lite, Jr., K.E., Morgan, D.S., and Collins, C.A. (2001). Groundwater hydrology of the upper Deschutes Basin, Oregon. U.S. Geological Survey Water Resources Investigations Report 00-4162, 74 p. Accessed February 24, 2021 from https://pubs.usgs.gov/wri/wri004162/	Approximated from Fig. 1 of Gannett et al. (2001) and influenced by the spatial distribution of groundwater wells recorded by the Oregon Water Resources Department (2021).
Fort Rock Basin	-	Miller, D.W. (1984). Appraisal of ground water conditions in the Fort Rock Basin, Lake County, Oregon. Oregon Water Resources Department Open File Report, 78 pp. Accessed February 24, 2021 from https://digital.osl.state.or.us/islandora/object/osl%3A13403/datastream/OBJ/view	Approximated from Fig. 1 of Miller (1984) and influenced by the spatial distribution of groundwater wells recorded by the Oregon Water Resources Department (2021).
Harney Basin	-	Albano, C., Minor, B., Freed, Z., Huntington, J.L. (2020). Status and Trends of Groundwater Dependent Vegetation in Relation to Climate and Shallow Groundwater in the Harney Basin, Oregon. Report for The Nature Conservancy in Oregon (Contract ORFO-090418-aa01), 63 pp.	Approximated from Fig. 1 of) and influenced by the spatial distribution of groundwater wells recorded by the Oregon Water Resources Department (2021).

Aquifer System	Broader System	References			Delineation steps and sources
			groundwater_planofstudy_Dec2016.pdf		
Lahontan Valley	-	Smith, D.W., Buto, S.G., Welborn, T.L. (2016). Groundwater level change and evaluation of simulated water levels for irrigated areas in Lahontan Valley, Churchill County, west-central Nevada, 1992–2012. U.S. Geological Survey Scientific Investigations Report 2016-5045, 23 pp. Accessed March 10, 2021 from https://pubs.usgs.gov/sir/2016/5045/sir20165045.pdf	-	-	Approximated from Fig. 4 of Smith et al. (2016)
Upper Reese River Valley in northern Nevada	-	Bredehoeft, J.D., Farvolden, R.N. (1963). International Association of Scientific Hydrology, Commission of Subterranean Waters, Publication no. 64, p. 197–212. Accessed March 23, 2021 from http://hydrologie.org/redbooks/a064/064047.pdf	-	-	Approximated from Fig. 1 of Bredehoeft and Farvolden (1963) and guided by locations of wells and topography (especially to distinguish upper and lower portions of basin)
Big Smoky Valley	-	Meinzer, O.E. (1916). Ground water in Big Smoky Valley, Nevada. Water Supply Paper 375-D. 34 pp. Accessed March 23, 2021 from https://pubs.usgs.gov/wsp/0375d/report.pdf	-	-	Approximated from Plate 6 (page 7 of 34 in document) of Meinzer (1916).
Lolo-Bitterroot Valley	Lolo-Bitterroot-Missoula Valleys	Birar, D.W., Dutton, D.M. (1999). Hydrogeology and Aquifer Sensitivity of the Bitterroot Valley, Ravalli County, Montana. Water Resources Investigations Report, 99-4219. Accessed March 2, 2021 from https://pubs.usgs.gov/wri/1999/4219/report.pdf	Smith, L. N., LaFave, J., I., Patton, T. W. (2013). Groundwater resources of the Lolo-Bitterroot area: Mineral, Missoula, and Ravalli Counties, Montana. Montana Bureau of Mines and Geology. Montana Groundwater Assessment Atlas No. 4. Accessed March 2, 2021 from http://www.mbmgt.mtech.edu/pdf-publications/gwaa4a.pdf	Smith, L. N. (2006). Hydrologic framework of the Lolo-Bitterroot Area groundwater characterization study. Montana Bureau of Mines and Geology. Montana Ground Water Assessment Atlas 4-B-02, 1 sheet, scale 1:250,000. Accessed March 2, 2021 from http://mbmg.mtech.edu/pdf-publications/GWAA04B-02.pdf	Approximated from Fig. 1 of Birar and Dutton (1999), Fig. 2 of Smith et al. (2003) and map by Smith (2006).
Missoula Valley	Lolo-Bitterroot-Missoula Valleys	Smith, L. N., LaFave, J., I., Patton, T. W. (2013). Groundwater resources of the Lolo-Bitterroot area: Mineral, Missoula, and Ravalli Counties, Montana. Montana Bureau of Mines and Geology. Montana Groundwater Assessment Atlas No. 4. Accessed March 2, 2021 from http://www.mbmgt.mtech.edu/pdf-publications/gwaa4a.pdf	-	-	Approximated from Fig. 2 of Smith et al. (2003).

Aquifer System	Broader System	References		Delineation steps and sources	
KalisPELL Valley	Flathead Valley	Wheaton, J., Rose, J., Bobet, A., Gebril, A. (2015). Flathead Valley Deep Aquifer: Geologic Setting and Hydrogeologic Implications. Montana Bureau of Mines and Geology presentation at the 2015 Clark Fork Symposium. Accessed March 2, 2021 from https://www.mbm-g.mtech.edu/gwip/gwip_pdf/2015/2015_ClarkForkSymposium-Flathead.pdf	LaFave, J.I. (2000). Potentiometric surface map of the Deep Aquifer, Kalispell valley, Flathead County, Montana. Montana Ground-Water Assessment Atlas GWA-02B-2. Accessed March 29, 2021 from https://ngmdb.usgs.gov/Prodesc/proddesc_61253.htm	-	Approximated from Figure shown in the upper right corner of the poster by Wheaton et al. (2015) and map by LaFave (2000).
Mission Valley and Irvine Flats	Flathead Valley	LaFave, J. I. (2004). Potentiometric Surface Map of the Southern Part of the Flathead Lake Area, Lake, Missoula, Sanders Counties, Montana. Montana Ground-Water Assessment Atlas No. 2, Part B, Map 4. Montana Bureau of Mines and Geology, A Department of Montana Tech of The University of Montana, Butte, MT.	Smith, L.N. (2004). Hydrogeologic framework of the southern part of the Flathead Lake Area, Flathead, Lake, Missoula, and Sanders counties, Montana. Montana Bureau of Mines and Geology Montana Ground-Water Assessment Atlas 2-B-10, 1 sheet, scale 1:300,000. Accessed March 2, 2021 from http://mbmg.mtech.edu/gwepmaps/gwa02map10untitled.pdf	-	Approximated from map by Smith (2004) and LaFave (2004).
Flathead Lake Perimeter	Flathead Valley	LaFave, J. I., Smith, L. N., Patton, T. W. (2004). Ground-Water Resources of the Flathead Lake Area: Flathead, Lake, Missoula, and Sanders Counties, Montana. Part A—Descriptive Overview and Water-Quality Data. Montana Bureau of Mines and Geology, Butte, MT. Montana Ground-Water Assessment Atlas 2. Accessed March 2, 2021 from http://mbmg.mtech.edu/pdf/GWA_2.pdf	-	-	Approximated from Fig. 2 of LaFave et al. (2004)
Smith Subarea	Flathead Valley	LaFave, J. I., Smith, L. N., Patton, T. W. (2004). Ground-Water Resources of the Flathead Lake Area: Flathead, Lake, Missoula, and Sanders Counties, Montana. Part A—Descriptive Overview and Water-Quality Data. Montana Bureau of Mines and Geology, Butte, MT. Montana Ground-Water Assessment Atlas 2. Accessed March 2, 2021 from http://mbmg.mtech.edu/pdf/GWA_2.pdf	-	-	Approximated from Fig. 2 of LaFave et al. (2004)
Little Bitterroot Valley	Flathead Valley	LaFave, J. I., Smith, L. N., Patton, T. W. (2004). Ground-Water Resources of the Flathead Lake Area: Flathead, Lake, Missoula, and Sanders Counties, Montana. Part A—Descriptive Overview and Water-Quality Data. Montana Bureau of Mines and Geology, Butte, MT.	Meinzer, O.E. (1916). Artesian water for irrigation in Little Bitterroot Valley, Montana. Water Supply Paper 400, 34 pp. Accessed April 7, 2021 from https://pubs.usgs.gov/wsp/0400b/report.pdf	-	Approximated from Fig. 2 of LaFave et al. (2004) and Fig. 1 of Meinzer (1916).

Aquifer System	Broader System	References	Delineation steps and sources
Camas Prairie Basin	Flathead Valley	Montana Ground-Water Assessment Atlas 2. Accessed March 2, 2021 from http://mbmg.mtech.edu/pdf/GWA_2.pdf LaFave, J. I., Smith, L. N., Patton, T. W. (2004). Ground-Water Resources of the Flathead-Lake Area: Flathead, Lake, Missoula, and Sanders Counties, Montana. Part A—Descriptive Overview and Water-Quality Data. Montana Bureau of Mines and Geology, Butte, MT. Montana Ground-Water Assessment Atlas 2. Accessed March 2, 2021 from http://mbmg.mtech.edu/pdf/GWA_2.pdf	Approximated from Fig. 2 of LaFave et al. (2004)
Santa Maria Basin and Nipomo Valley	-	Hughes, J. L. (1977). Evaluation of Ground-water Quality in the Santa Maria Valley, California. US Geological Survey, Water Resources Division Report 76-128, 77 pp. Accessed March 7, 2021 from https://pubs.usgs.gov/wri/1976/0128/report.pdf	Approximated from Fig. 1 of Hughes (1977)
Lower Santa Ynez Valley	Santa Ynez Valley	Upson, J. E., Thomasson, H. G. (1951). Geology and water resources of the Santa Ynez river basin, Santa Barbara County, California (Vol. 2). US Geological Survey Water Supply Report 1102. Accessed March 7, 2021 from https://pubs.usgs.gov/wsp/1102/report.pdf	Approximated from pp. 22 map in Hamlin (1985) and Fig. 2 of Wilson (1950); Lower portion of Valley based on the delineated subareas: "Lompoc" and "Santa Rita" and "Buellton" subarea
Upper Santa Ynez Valley	Santa Ynez Valley	Upson, J. E., Thomasson, H. G. (1951). Geology and water resources of the Santa Ynez river basin, Santa Barbara County, California (Vol. 2). US Geological Survey Water Supply Report 1102. Accessed March 7, 2021 from https://pubs.usgs.gov/wsp/1102/report.pdf	Approximated from Fig. 2 of Wilson (1950); Upper portion of Valley based on the delineated subareas: "Santa Ynez" and "Headwater Subarea"
Cuyama Valley	-	Everett, R.R., Gibbs, D.R., Hanson, R.T., Sweetkind, D.S., Brandt, J.T., Falk, S.E., Harich, C.R. (2013). Geology, water-quality, hydrology, and geomechanics of the Cuyama Valley groundwater basin, California, 2008-12. U.S. Geological	Approximated from Fig. 1 of Everett et al. (2013) and Fig. 2 of Sweet

Aquifer System	Broader System	References	Delineation steps and sources	
		Survey Scientific Investigations Report 2013-5108, 62 pp. Accessed March 7, 2021 from https://pubs.usgs.gov/sir/2013/5108/pdf/sir2013-5108.pdf	Investigations Report 2013-5127, 58 pp. Accessed March 7, 2021 from https://pubs.usgs.gov/sir/2013/5127/pdf/sir2013-5127.pdf	
San Antonio Creek Valley	-	Martin, P. (1984). Development and calibration of a two-dimensional digital model for the analysis of the ground-water flow system in the San Antonio Creek Valley, Santa Barbara County, California. Water Resources Investigations Report 84-4340, 73 pp. Accessed March 7, 2021 from https://pubs.usgs.gov/wri/1984/4340/report.pdf	-	Approximated from maps on pages 8 and 9 of Martin (1984)
Beaver Valley	-	Sandberg, G. W. (1964). Ground-water resources of selected basins in southwestern Utah. US Geological Survey Open-File Report 66 pp. Accessed March 7, 2021 from https://ciwedd.org/wp-content/uploads/2018/09/1966-tech-pub-13-ground-water-resources-of-selected-basins-in-southwestern-utah-by-g-w-sandberg-usgs.pdf	Lee, W. T. (1908). Water resources of Beaver Valley, Utah. U.S. Geological Survey Water Supply Paper 217, 60 pp. Accessed April 6, 2021 from https://pubs.usgs.gov/wsp/0217/report.pdf	Approximated from the map on page 49 of Sandberg (1966) and plate 1 by Lee (1908)
Parowan Valley	-	Marston, T. M. (2017). Water resources of Parowan Valley, Iron County, Utah. U.S. Geological Survey Scientific Investigations Report 2017-5033, 45 pp. Accessed March 7 2021 from https://doi.org/10.3133/sir20175033 .	-	Approximated from Fig. 1 of Marston et al. (2017)
Cedar Valley	-	Brooks, L. E., Mason, J. L. (2005). Hydrology and simulation of ground-water flow in Cedar Valley, Iron County, Utah. (No. 2005-5170). U.S. Geological Survey Scientific Investigations Report 2005-5170, 127 pp. Accessed March 7, 2021 from https://pubs.usgs.gov/sir/2005/5170/PDF/SIR2005_5170.pdf	-	Approximated from Fig. 1 of Brooks et al. (2005)
Pahvant Valley	-	Holmes, W. F., Thiros, S. A. (1990). Ground-water hydrology of Pahvant Valley and adjacent areas, Utah. United States Geological Survey Technical Publication 98, 64 pp. Accessed March 7, 2021 from https://waterights.utah.gov/docSys/v920/y920/y9200006.pdf	-	Approximated from Fig. 8 of Brooks et al. (2005)

Aquifer System	Broader System	References			Delineation steps and sources
Central-Sevier Valley	-	Young, R. A., Carpenter, C. H. (1965). Ground-water conditions and storage in the Central-Sevier Valley, Utah. US Geological Survey Water Supply Paper 1787, 106 pp. Accessed March 7, 2021 from https://pubs.usgs.gov/wsp/1787/report.pdf	-	-	Approximated from Plate 1 of Young and Carpenter (1965)
Sanpete Valley	-	Richardson, C. B. (1907). Underground water in Sanpete and Central-Sevier Valleys, Utah. U.S. Geological Survey Water Supply and Irrigation Paper No. 199, 77 pp. Accessed March 7, 2021 from https://pubs.usgs.gov/wsp/0199/report.pdf	-	-	Approximated from Fig. 1 of Richardson (1907)
Juab Valley	-	Thiros, S. A., Stolp, B. J., Hadley, H. K., Steiger, J. I. (1996). Hydrology and simulation of ground-water flow in Juab Valley, Juab County, Utah. Utah Department of Natural Resources, Division of Water Rights Technical Report. 100 pp. Accessed March 7, 2021 from https://waterrights.utah.gov/docSys/v920/y920/y920000.pdf	-	-	Approximated from Fig. 1 of Thiros et al. (1996)
Milford-Blackrock Subarea	Escalante Valley	Sandberg, C. W. (1966). Ground-water resources of selected basins in southwestern Utah. US Geological Survey Open File Report. 66 pp. Accessed March 7, 2021 from https://ciwcd.org/wp-content/uploads/2018/09/1966-tech-pub-13-ground-water-resources-of-selected-basins-in-southwestern-utah-by-g-w-sandberg-usgs.pdf	Fix, P. F., Nelson, W. B., Lofgren, B. E., Butler, R. G. (1950). Ground water in the Escalante Valley, Beaver, Iron, and Washington Counties, Utah. Technical Publication 6-102 pp. Accessed March 7, 2021 from https://waterrights.utah.gov/docSys/v920/w920/w9200085.pdf	-	Approximated from Fig. 1 of Fix et al. (1950) with subareas ("Districts") approximated from the map on page 49 of Sandberg (1966).
Beryl-Enterprise Subarea	Escalante Valley	Sandberg, C. W. (1964). Ground-water resources of selected basins in southwestern Utah. US Geological Survey Open File Report. 66 pp. Accessed March 7, 2021 from https://ciwcd.org/wp-content/uploads/2018/09/1966-tech-pub-13-ground-water-resources-of-selected-basins-in-southwestern-utah-by-g-w-sandberg-usgs.pdf	Fix, P. F., Nelson, W. B., Lofgren, B. E., Butler, R. G. (1950). Ground water in the Escalante Valley, Beaver, Iron, and Washington Counties, Utah. Technical Publication 6-102 pp. Accessed March 7, 2021 from https://waterrights.utah.gov/docSys/v920/w920/w9200085.pdf	-	Approximated from Fig. 1 of Fix et al. (1950) with subareas ("Districts") approximated from the map on page 49 of Sandberg (1966).
Lund Subarea	Escalante Valley	Sandberg, C. W. (1964). Ground-water resources of selected basins in	Fix, P. F., Nelson, W. B., Lofgren, B. E., Butler, R. G. (1950). Ground	-	Approximated from Fig. 1 of Fix et al. (1950) with subareas

Aquifer System	Broader System	References			Delineation steps and sources
		southwestern Utah. US Geological Survey Open File Report. 66 pp. Accessed March 7, 2024 from https://ciowcd.org/wp-content/uploads/2018/00/1966-tech-pub-13-ground-water-resources-of-selected-basins-in-southwestern-utah-by-g-w-sandburg-usgs.pdf	water in the Escalante Valley, Beaver, Iron, and Washington Counties, Utah Technical Publication 6-102 pp. Accessed March 7, 2024 from https://waterrights.utah.gov/docSys/v920/w920/w9200085.pdf		("Districts") approximated from the map on page 49 of Sandberg (1966).
Rush Valley	-	Gardner, P. M., Kirby, S. (2011). Hydrogeologic and geochemical characterization of groundwater resources in Rush Valley, Tooele County, Utah. U. S. Geological Survey Scientific Investigations Report 2011-5068, 80 pp. Accessed March 7, 2024 from https://pubs.usgs.gov/sir/2011/5068/pdf/sir20115068.pdf	-	-	Approximated from Fig. 1 of Gardner and Kirby (2011).
Tooele Valley	-	Thomas, H. E. (1946). Ground water in Tooele Valley, Tooele County, Utah (No. 4, pp. 91-238). Utah Department of Natural Resources, Division of Water Rights. 148 pp. Accessed March 7, 2024 from https://waterrights.utah.gov/docSys/v920/w920/w9200083.pdf	-	-	Approximated from Fig. 1 of Gardner and Kirby (2011).
Goshen Valley	-	Brooks, L.E. (2013). Evaluation of the groundwater flow model for southern Utah and Goshen Valleys, Utah, updated to conditions through 2011, with new projections and groundwater management simulations. U.S. Geological Survey Open File Report 2013-1171, 35 pp. Accessed March 7, 2024 from https://pubs.usgs.gov/of/2013/1171/pdf/ofr2013-1171.pdf	-	-	Approximated from Fig. 1 of Brooks (2013)
Southern Utah Valley	-	Brooks, L.E. (2013). Evaluation of the groundwater flow model for southern Utah and Goshen Valleys, Utah, updated to conditions through 2011, with new projections and groundwater management simulations. U.S. Geological Survey Open File Report 2013-1171, 35 pp. Accessed March 7, 2024 from https://pubs.usgs.gov/of/2013/1171/pdf/ofr2013-1171.pdf	Selck, B. J., Carling, G. T., Kirby, S. M., Hansen, N. C., Bickmore, B. R., Tingey, D. G., Rey, K., Wallace, J. Jordan, J. L. (2018). Investigating anthropogenic and geogenic sources of groundwater contamination in a semi-arid alluvial basin, Goshen Valley, UT, USA. Water, Air, Soil Pollution, 229(6), 1-17.	-	Approximated from Fig. 1 of Brooks (2013)

Aquifer System	Broader System	References			Delineation steps and sources
Cedar Valley near Utah Lake	-	Jordan, J. L. (2013). Aquifer Parameter Estimation from Aquifer Tests and Specific-capacity Data in Cedar Valley and the Cedar Pass Area, Utah County, Utah. Utah Geological Survey Special Study 146.	-	-	Approximated from Fig. 4 of Jordan (2013)
Sevier Desert	-	Mower, R. W., Feltis, R. D. (1968). Ground-water hydrology of the Sevier Desert, Utah. U.S. Geological Survey, Water Supply Paper 1854, 88 pp. Accessed March 7, 2021 from https://pubs.usgs.gov/wsp/1854/report.pdf	-	-	Approximated from Fig. 4 of Mower and Feltis (1968)
Ashley Valley	Uinta Basin	Hood, J. W. (1976). Characteristics of aquifers in the northern Uinta Basin area, Utah and Colorado. U.S. Geological Survey and Utah Department of Natural Resources, Division of Water Rights Technical Publication 53, 71 pp. Accessed March 7, 2021 from https://waterrights.utah.gov/docSys/v920/w920/w920009f.pdf	Zhang, Y., Gable, C. W., Zylowski, G. A., Walter, L. M. (2009). Hydrogeochemistry and gas compositions of the Uinta Basin: A regional-scale overview. AAPG bulletin, 93(8), 1087-1118.	Hood, J. W. (1977). Hydrologic Evaluation of Ashley Valley, Northern Uinta Basin Area, Utah (Vol. 76, No. 496). U.S. Geological Survey and Utah Department of Natural Resources, Division of Water Rights Technical Publication 54, 71 pp. Accessed March 7, 2021 from https://pubs.usgs.gov/unnumbered/770043723/report.pdf	Northern Uinta Basin outline from Plates 1-3 of Hood (1976). Ashley Valley outline approximated from Fig. 1 of Hood (1977) and extended to northern margin of Uinta Basin
Birds Nest Aquifer	Uinta Basin	Vanden Berg, M. D., Lehle, D. R., Carney, S. M., Morgan, C. D. (2013). Geological characterization of the Bird's Nest Aquifer, Uinta Basin, Utah: assessment of the aquifer's potential as a saline water disposal unit. Utah Geological Survey Special Study 147, 57 pp. Accessed March 9, 2021 from https://ugspub.nr.utah.gov/publications/special_studies/ss-147/ss-147.pdf	-	-	Birds Nest Aquifer (lower) outline approximated from Fig. 19 of Vanden Berg et al. (2013).
Roosevelt Valley and Uinta River Basin and Pelican Lake region	Uinta Basin	Hood, J. W. (1976). Characteristics of aquifers in the northern Uinta Basin area, Utah and Colorado. U.S. Geological Survey and Utah Department of Natural Resources, Division of Water Rights Technical Publication 53, 71 pp. Accessed March 7, 2021 from https://waterrights.utah.gov/docSys/v920/w920/w920009f.pdf	Lambert, P. M., Marston, T., Kimball, B. A., Stolp, B. J. (2011). Assessment of groundwater/surface-water interaction and simulation of potential streamflow depletion induced by groundwater withdrawal, Uinta River near Roosevelt, Utah: U.S. Geological Survey Scientific Investigations Report 2011-5044, 47 pp. Accessed March 9, 2021 from https://pubs.usgs.gov/sir/2011/5044/pdf/sir20115044.pdf	-	Northern Uinta Basin and Roosevelt Valley, Uinta River Basin and Pelican Lake region outlines approximated from Plates 1-3 of Hood (1976). Further information on Roosevelt Valley available via Lambert et al. (2011)

Aquifer System	Broader System	References			Delineation steps and sources
Eastern Uinta Basin and Diamond Plateau	Uinta Basin	Hood, J. W. (1976). Characteristics of aquifers in the northern Uinta Basin area, Utah and Colorado. U.S. Geological Survey and Utah Department of Natural Resources, Division of Water Rights Technical Publication 53, 71 pp. Accessed March 7, 2021 from https://waterrights.utah.gov/docSys/v920/w920/w920009f.pdf	-	-	Northern Uinta Basin and Diamond Plateau region approximated from Plates 1-3 of Hood (1976).
Western Uinta Basin	Uinta Basin	Hood, J. W. (1976). Characteristics of aquifers in the northern Uinta Basin area, Utah and Colorado. U.S. Geological Survey and Utah Department of Natural Resources, Division of Water Rights Technical Publication 53, 71 pp. Accessed March 7, 2021 from https://waterrights.utah.gov/docSys/v920/w920/w920009f.pdf	-	-	Northern Uinta Basin approximated from Plates 1-3 of Hood (1976).
Spring Valley	-	Gardner, P. M., Masbruch, M. D. (2015). Hydrogeologic and geochemical characterization of groundwater resources in Deep Creek Valley and adjacent areas, Juab and Tooele Counties, Utah, and Elko and White Pine Counties, Nevada. US Geological Survey Scientific Investigations Report 2015-5097, 66 pp. Accessed March 10, 2021 from https://pubs.usgs.gov/sir/2015/5097/sir20155097.pdf	-	-	Approximated from Fig. 6 of Gardner and Masbruch (2015)
Snake Valley	-	Gardner, P. M., Masbruch, M. D. (2015). Hydrogeologic and geochemical characterization of groundwater resources in Deep Creek Valley and adjacent areas, Juab and Tooele Counties, Utah, and Elko and White Pine Counties, Nevada. US Geological Survey Scientific Investigations Report 2015-5097, 66 pp. Accessed March 10, 2021 from https://pubs.usgs.gov/sir/2015/5097/sir20155097.pdf	-	-	Approximated from Fig. 6 of Gardner and Masbruch (2015)
Deep Creek Valley	-	Gardner, P. M., Masbruch, M. D. (2015). Hydrogeologic and geochemical characterization of groundwater resources in Deep Creek Valley and adjacent areas, Juab and Tooele Counties, Utah, and Elko and White Pine	-	-	Approximated from Fig. 6 of Gardner and Masbruch (2015)

Aquifer System	Broader System	References			Delineation steps and sources
		Counties, Nevada. US Geological Survey Scientific Investigations Report 2015-5007, 66 pp. Accessed March 10, 2021 from https://pubs.usgs.gov/sir/2015/5007/sir20155007.pdf			
Tippett and Antelope Valleys	-	Gardner, P. M., Masbruch, M. D. (2015). Hydrogeologic and geochemical characterization of groundwater resources in Deep Creek Valley and adjacent areas, Juab and Tooele Counties, Utah, and Elko and White Pine Counties, Nevada. US Geological Survey Scientific Investigations Report 2015-5007, 66 pp. Accessed March 10, 2021 from https://pubs.usgs.gov/sir/2015/5007/sir20155007.pdf	-	-	Approximated from Fig. 6 of Gardner and Masbruch (2015)
Lake Valley	-	Hurlow, H. A. (2014). Hydrogeologic studies and groundwater monitoring in Snake Valley and adjacent hydrographic areas, west-central Utah and east-central Nevada (Vol. 135). Utah Geological Survey Bulletin 135, 272 pp.	-	-	Approximated from map on page 6 of Hurlow (2014)
Hamlin Valley	-	Hurlow, H. A. (2014). Hydrogeologic studies and groundwater monitoring in Snake Valley and adjacent hydrographic areas, west-central Utah and east-central Nevada (Vol. 135). Utah Geological Survey Bulletin 135, 272 pp.	-	-	Approximated from map on page 6 of Hurlow (2014)
Dry Lake Valley and Dreamer Valley	-	Hurlow, H. A. (2014). Hydrogeologic studies and groundwater monitoring in Snake Valley and adjacent hydrographic areas, west-central Utah and east-central Nevada (Vol. 135). Utah Geological Survey Bulletin 135, 272 pp.	-	-	Approximated from map on page 6 of Hurlow (2014)
White River Valley	-	Maxey, G. B., Eakin, T. E. (1949). Ground water in White River Valley, White Pine, Nye, and Lincoln Counties, Nevada. U.S. Department of the Interior Water Resources Bulletin 8, 64 pp. Accessed March 10, 2021 from https://www.nre.gov/docs/ML0331/ML033140348.pdf	-	-	Approximated from maps on page 63 and 64 of Maxey and Eakin (1949)
Step toe Valley	-	Frick, E. (1985). Quantitative analysis of groundwater flow in valley-fill deposits in Step toe Valley, Nevada. Doctoral dissertation, University of Nevada, Reno.	-	-	Approximated from Fig. 1 of Frick (1985)

Aquifer System	Broader System	References	Delineation steps and sources
		199 pp. Accessed March 10, 2021 from https://scholarworks.unr.edu/bitstream/handle/11714/1293/Mackay210_Frick.pdf?sequence=1	
Railroad Valley	-	Rose, T. P., Davison, M. L., Smith, D. K., Kenneally, J. M. (1989). Isotope hydrology investigation of regional groundwater flow in central Nevada. Hydrologic Resources Management Program and Underground Test Area Operable Unit F.Y. 1997 Progress Report, Chapter 6. Accessed March 10, 2021 from https://core.ac.uk/download/pdf/204554577.pdf#page=62	Approximated from Fig. 3 of Rose et al. (1989)
Pahranagat Valley	-	Burbey, T. J. (1997). Hydrogeology and potential for ground-water development, carbonate-rock aquifers in southern Nevada and southeastern California. US Geological Survey Water Resources Investigations Report 95-4168, 70 pp. Accessed March 10, 2021 from https://pubs.usgs.gov/wri/1995/4168/report.pdf	Approximated from Fig. 4 of Burbey (1997)
Coyote Springs Valley	-	Burbey, T. J. (1997). Hydrogeology and potential for ground-water development, carbonate-rock aquifers in southern Nevada and southeastern California. US Geological Survey Water Resources Investigations Report 95-4168, 70 pp. Accessed March 10, 2021 from https://pubs.usgs.gov/wri/1995/4168/report.pdf	Approximated from Fig. 7 of Burbey (1997)
Las Vegas Basin	-	Burbey, T. J. (1997). Hydrogeology and potential for ground-water development, carbonate-rock aquifers in southern Nevada and southeastern California. US Geological Survey Water Resources Investigations Report 95-4168, 70 pp. Accessed March 10, 2021 from https://pubs.usgs.gov/wri/1995/4168/report.pdf	Approximated from Fig. 10 of Burbey (1997)
Amargosa Desert	-	Burbey, T. J. (1997). Hydrogeology and potential for ground-water development, carbonate-rock aquifers in southern Nevada and southeastern California. US Geological Survey Water Resources Investigations Report 95-4168, 70 pp.	Approximated from Fig. 14 of Burbey (1997)

Aquifer System	Broader System	References	Delineation steps and sources
		Accessed March 10, 2021 from https://pubs.usgs.gov/wri/1995/4168/report.pdf	
Pahrump Valley	-	Burbey, T. J. (1997). Hydrogeology and potential for ground-water development, carbonate-rock aquifers in southern Nevada and southeastern California. US Geological Survey Water Resources Investigations Report 95-4168, 70 pp. Accessed March 10, 2021 from https://pubs.usgs.gov/wri/1995/4168/report.pdf	Approximated from Fig. 15 of Burbey (1997)
Mesquite Valley	-	Burbey, T. J. (1997). Hydrogeology and potential for ground-water development, carbonate-rock aquifers in southern Nevada and southeastern California. US Geological Survey Water Resources Investigations Report 95-4168, 70 pp. Accessed March 10, 2021 from https://pubs.usgs.gov/wri/1995/4168/report.pdf	Approximated from Fig. 16 of Burbey (1997)
Indian Springs Valley	-	Burbey, T. J. (1997). Hydrogeology and potential for ground-water development, carbonate-rock aquifers in southern Nevada and southeastern California. US Geological Survey Water Resources Investigations Report 95-4168, 70 pp. Accessed March 10, 2021 from https://pubs.usgs.gov/wri/1995/4168/report.pdf	Approximated from Fig. 13 of Burbey (1997)
Three Lakes Valley	-	Burbey, T. J. (1997). Hydrogeology and potential for ground-water development, carbonate-rock aquifers in southern Nevada and southeastern California. US Geological Survey Water Resources Investigations Report 95-4168, 70 pp. Accessed March 10, 2021 from https://pubs.usgs.gov/wri/1995/4168/report.pdf	Approximated from Fig. 12 of Burbey (1997)
Churchill Valley	Carson River Basin	Maurer, D. K. (2011). Geologic framework and hydrogeology of the middle Carson River Basin, Eagle, Dayton, and Churchill Valleys, West-Central Nevada. U. S. Geological Survey Scientific Investigations Report 2011-5055, 74 pp. Accessed March 10, 2021 from	Approximated from Fig. 4 of Maurer et al. (2011).

Aquifer System	Broader System	References			Delineation steps and sources
		https://pubs.usgs.gov/sir/2011/5055/pdf/sir20115055.pdf			
Dayton Valley	Carson River Basin	Maurer, D. K. (2011). Geologic framework and hydrogeology of the middle Carson River Basin, Eagle, Dayton, and Churchill Valleys, West-Central Nevada. U. S. Geological Survey Scientific Investigations Report 2011-5055. 74 pp. Accessed March 10, 2021 from https://pubs.usgs.gov/sir/2011/5055/pdf/sir20115055.pdf	-	-	Approximated from Fig. 1 of Maurer et al. (2011).
Carson Valley	Carson River Basin	Maurer, D. K. (2011). Geologic framework and hydrogeology of the middle Carson River Basin, Eagle, Dayton, and Churchill Valleys, West-Central Nevada. U. S. Geological Survey Scientific Investigations Report 2011-5055. 74 pp. Accessed March 10, 2021 from https://pubs.usgs.gov/sir/2011/5055/pdf/sir20115055.pdf	-	-	Approximated from Fig. 1 of Maurer et al. (2011).
Eagle Valley	Carson River Basin	Maurer, D. K. (2011). Geologic framework and hydrogeology of the middle Carson River Basin, Eagle, Dayton, and Churchill Valleys, West-Central Nevada. U. S. Geological Survey Scientific Investigations Report 2011-5055. 74 pp. Accessed March 10, 2021 from https://pubs.usgs.gov/sir/2011/5055/pdf/sir20115055.pdf	Maurer, D.K., Thodal, C.E. (2000). Quantity and chemical quality of recharge, and updated water budgets, for the basin-fill aquifer in Eagle Valley, western Nevada. U.S. Geological Survey Water Resources Investigations Report 00-4289. 62 pp. Accessed November 29, 2021 from https://pubs.usgs.gov/wri/1999/4289/report.pdf	-	Approximated from Fig. 1 of Maurer et al. (2011) and Fig. 1 by Maurer and Thodal (2000)
Abbotsford-Sumas Aquifer	Puget Sound Lowland	Seibek, J., Allen, D. M. (2005). Numerical groundwater flow model of the Abbotsford-Sumas aquifer, central Fraser Lowland of BC, Canada, and Washington State, US. Report prepared for Environment Canada, Vancouver, 203 (accessed February 9, 2021 via https://www.sfu.ca/personal/dallen/AB_Modeling_Report_Final.pdf)	Cox, S. E., Kahle, S. C. (1999). Hydrogeology, ground-water quality, and sources of nitrate in lowland glacial aquifers of Whatcom County, Washington, and British Columbia, Canada (Vol. 98, No. 4195). US Department of the Interior, US Geological Survey. https://pubs.er.usgs.gov/publication/wri984195	Vaccaro, J. J., Hansen, A. J., Jones, M. A. (1998). Hydrogeologic framework of the Puget Sound aquifer system, Washington and British Columbia. U.S. Geological Survey Professional Paper 1424-D, 87 pp. Accessed March 11, 2021 from https://pubs.usgs.gov/pp/1424d/report.pdf	Broader Puget Sound Lowland approximated from Vaccaro et al. (1998). Abbotsford-Sumas aquifer approximated from Map 4 of Seibek and Allen (2005) and Plate 2 of Cox and Kahle (1999) (the latter reference led to our extending the southern boundary of the aquifer ~40 km farther south than delineated by Seibek and Allen (2005))
Strait of Juan de Fuca Sub-basin	Puget Sound Lowland	Vaccaro, J. J., Hansen, A. J., Jones, M. A. (1998). Hydrogeologic framework of the Puget Sound aquifer system,	Washington Department of Ecology (2010). Puget Sound Groundwater Toxics Loading Analysis: Direct	-	Broader Puget Sound Lowland approximated from Vaccaro et al. (1998). Sub-basin outline

Aquifer System	Broader System	References		Delineation steps and sources	
		Washington and British Columbia. U.S. Geological Survey Professional Paper 1424-D, 87 pp. Accessed March 11, 2021 from https://pubs.usgs.gov/pp/1424d/report.pdf	Discharge Pathway. Publication No. 10-03-122, 26 pp. Accessed March 12, 2021 from https://apps.ecology.wa.gov/publications/documents/1003122.pdf	approximated from Fig. 2 of Washington Department of Ecology (2010).	
Hood Canal and Admiralty Inlet Sub-basin	Puget Sound Lowland	Vaccaro, J. J., Hansen, A. J., Jones, M. A. (1998). Hydrogeologic framework of the Puget Sound aquifer system, Washington and British Columbia. U.S. Geological Survey Professional Paper 1424-D, 87 pp. Accessed March 11, 2021 from https://pubs.usgs.gov/pp/1424d/report.pdf	Washington Department of Ecology (2010). Puget Sound Groundwater Toxics Loading Analysis: Direct Discharge Pathway. Publication No. 10-03-122, 26 pp. Accessed March 12, 2021 from https://apps.ecology.wa.gov/publications/documents/1003122.pdf	-	Broader Puget Sound Lowland approximated from Vaccaro et al. (1998). Sub-basin outline approximated from Fig. 2 of Washington Department of Ecology (2010).
South Sound and Commencement Bay	Puget Sound Lowland	Vaccaro, J. J., Hansen, A. J., Jones, M. A. (1998). Hydrogeologic framework of the Puget Sound aquifer system, Washington and British Columbia. U.S. Geological Survey Professional Paper 1424-D, 87 pp. Accessed March 11, 2021 from https://pubs.usgs.gov/pp/1424d/report.pdf	Washington Department of Ecology (2010). Puget Sound Groundwater Toxics Loading Analysis: Direct Discharge Pathway. Publication No. 10-03-122, 26 pp. Accessed March 12, 2021 from https://apps.ecology.wa.gov/publications/documents/1003122.pdf	-	Broader Puget Sound Lowland approximated from Vaccaro et al. (1998). Sub-basin outline approximated from Fig. 2 of Washington Department of Ecology (2010).
Main Basin and Sinclair Inlet and Western Hood Canal	Puget Sound Lowland	Vaccaro, J. J., Hansen, A. J., Jones, M. A. (1998). Hydrogeologic framework of the Puget Sound aquifer system, Washington and British Columbia. U.S. Geological Survey Professional Paper 1424-D, 87 pp. Accessed March 11, 2021 from https://pubs.usgs.gov/pp/1424d/report.pdf	Washington Department of Ecology (2010). Puget Sound Groundwater Toxics Loading Analysis: Direct Discharge Pathway. Publication No. 10-03-122, 26 pp. Accessed March 12, 2021 from https://apps.ecology.wa.gov/publications/documents/1003122.pdf	-	Broader Puget Sound Lowland approximated from Vaccaro et al. (1998). Sub-basin outline approximated from Fig. 2 of Washington Department of Ecology (2010).
Whidbey and Port Gardner Basins	Puget Sound Lowland	Vaccaro, J. J., Hansen, A. J., Jones, M. A. (1998). Hydrogeologic framework of the Puget Sound aquifer system, Washington and British Columbia. U.S. Geological Survey Professional Paper 1424-D, 87 pp. Accessed March 11, 2021 from https://pubs.usgs.gov/pp/1424d/report.pdf	Washington Department of Ecology (2010). Puget Sound Groundwater Toxics Loading Analysis: Direct Discharge Pathway. Publication No. 10-03-122, 26 pp. Accessed March 12, 2021 from https://apps.ecology.wa.gov/publications/documents/1003122.pdf	-	Broader Puget Sound Lowland approximated from Vaccaro et al. (1998). Sub-basin outline approximated from Fig. 2 of Washington Department of Ecology (2010).
Strait of Georgia Sub-basin	Puget Sound Lowland	Vaccaro, J. J., Hansen, A. J., Jones, M. A. (1998). Hydrogeologic framework of the Puget Sound aquifer system, Washington and British Columbia. U.S. Geological Survey Professional Paper 1424-D, 87 pp. Accessed March 11, 2021 from https://pubs.usgs.gov/pp/1424d/report.pdf	Washington Department of Ecology (2010). Puget Sound Groundwater Toxics Loading Analysis: Direct Discharge Pathway. Publication No. 10-03-122, 26 pp. Accessed March 12, 2021 from https://apps.ecology.wa.gov/publications/documents/1003122.pdf	-	Broader Puget Sound Lowland approximated from Vaccaro et al. (1998). Sub-basin outline approximated from Fig. 2 of Washington Department of Ecology (2010).

Aquifer System	Broader System	References	Delineation steps and sources
		2021 from https://pubs.usgs.gov/pp/1424d/report.pdf	https://apps.ecology.wa.gov/publications/documents/1003122.pdf
Fraser River Delta	Puget Sound Lowland	Vaccaro, J. J., Hansen, A. J., Jones, M. A. (1998). Hydrogeologic framework of the Puget Sound aquifer system, Washington and British Columbia. U.S. Geological Survey Professional Paper 1424-D, 87 pp. Accessed March 11, 2021 from https://pubs.usgs.gov/pp/1424d/report.pdf	Bridger, D. W., Allen, D. M. (2006). An investigation into the effects of diffusion on salinity distribution beneath the Fraser River Delta, Canada. Hydrogeology Journal , 14(8), 1423.
Lower Fraser Valley	Puget Sound Lowland	Vaccaro, J. J., Hansen, A. J., Jones, M. A. (1998). Hydrogeologic framework of the Puget Sound aquifer system, Washington and British Columbia. U.S. Geological Survey Professional Paper 1424-D, 87 pp. Accessed March 11, 2021 from https://pubs.usgs.gov/pp/1424d/report.pdf	Wilson, J. E., Brown, S., Schreier, H., Scovill, D., Zobel, M. (2008). Arsenic in groundwater wells in Quaternary deposits in the Lower Fraser Valley of British Columbia. Canadian Water Resources Journal , 33(4), 397-412.
Big Chino Valley	-	Langenheim, V. E., Duval, J. S., Wirt, L., DeWitt, E. (2000). Preliminary report on geophysics of the Verde River headwaters region, Arizona. U.S. Geological Survey Open-File Report 00-403, 28 pp. Accessed March 12, 2021 from https://pubs.usgs.gov/of/2000/0403/pdf/of00-403p.pdf	-
Little Chino Valley	-	Kennedy, J.R., Kahler, L.M., Read, A.L. (2019). Aquifer storage change and storage properties, 2010-2017, in the Big Chino Subbasin, Yavapai County, Arizona. U.S. Geological Survey Scientific Investigations Report 2019-5060, 39 pp. Accessed March 12, 2021 from https://doi.org/10.3133/sir20195060	-
Lonesome and Prescott Valleys	-	Matlock, W. G., Davis, P. R., Roth, R. L. (1973). Groundwater in Little Chino Valley, Arizona. Tucson, University of Arizona, College of Agriculture, Agricultural Experiment Station. Technical Bulletin, 201, 19. Accessed March 12, 2021 from https://repository.arizona.edu/bitstream/handle/10150/602177/TB178.pdf?sequence=1	-
			Approximated from map on pp. 12 of Langenheim et al. (2000) and Fig. 2 by Kennedy et al. (2019)
			Approximated from Fig. 4 of Kennedy et al. (2019)
			Approximated from Plate 4 of Matlock et al. (1973)

Aquifer System	Broader System	References		Delineation steps and sources	
Santa Rosa Valley	-	Valin, Z.C., McLaughlin, R. J. (2005). Locations and data for water wells of the Santa Rosa Valley, Sonoma County, California. U.S. Geological Survey Open File Report 2005-1318, 16 pp. Accessed March 12, 2021 from https://pubs.usgs.gov/of/2005/1318/of2005-1318.pdf	Cardwell, G. T. (1958). Geology and ground water in the Santa Rosa and Petaluma Valley areas, Sonoma County, California. U.S. Geological Survey Water Supply Paper 1427, 284 pp. Accessed November 29, 2021 from https://pubs.usgs.gov/wsp/1427/report.pdf	-	Approximated from map on page 2 of Valin and McLaughlin (2005) and Fig. 1 by Cardwell (1958)
Big Bear Valley	-	Flint, L. E., Brandt, J., Christensen, A. H., Flint, A.L., Hevesi, J.A., Jachens, R., Kulongoski, J.T., Martin, P., Sneed, M. (2012). Geohydrology of Big Bear Valley, California: phase 1—Geologic framework, recharge, and preliminary assessment of the source and age of groundwater. US Geological Survey Scientific Investigations Report 2012-5100, 122 pp. Accessed March 14, 2021 from https://pubs.usgs.gov/sir/2012/5100/pdf/sir20125100.pdf	-	-	Approximated from Fig. 4 of Flint et al. (2012)
Grand Forks Aquifer	-	Wei, M., Allen, D. M., Carmichael, V., Ronneseth, K. (2010). State of understanding of the hydrogeology of the Grand Forks aquifer. Water Stewardship Division, BC Ministry of Environment Report, 99 pp. Accessed March 15, 2021 from https://www.grandforks.ca/wp-content/uploads/reports/2010-Hydrogeology-Study-of-Grand-Forks-area.pdf	-	-	Approximated from Fig. 2 of Wei et al. (2010)
Kettle River Valley	-	Walters, K. L. (1960). Availability of ground water at the border stations at Laurier and Ferry, Washington. US Geological Circular 422, 12 pp. Accessed March 15, 2021 from https://pubs.usgs.gov/of/1960/0422/report.pdf	-	-	Approximated from Fig. 4 of Walters (1960); British Columbia portion estimated based on well completion data
Lower Colville Basin	Colville Basin	Kahle, S.C., Longpré, C.J., Smith, R.R., Sumioka, S.S., Watkins, A.M., Kresch, D.L. (2003). Water Resources of the Ground Water System in the Unconsolidated Deposits of the Colville River Watershed, Stevens County, Washington. U.S. Geological Survey Water Resources Investigations Report 03-4128, 84 pp. Accessed March 15, 2021 from	-	-	Approximated from Fig. 6 of Kahle et al. (2003) with guidance from well completion report data. Upper and Lower basins defined near the town of Addy (Washington).

Aquifer System	Broader System	References			Delineation steps and sources
		https://pubs.usgs.gov/wri/wri034128/pdf/wri034128.pdf			
Upper Colville Basin	Colville Basin	Kahle, S.C., Longpré, C.I., Smith, R.R., Sumioka, S.S., Watkins, A.M., Kresch, D.L. (2003). Water Resources of the Ground-Water System in the Unconsolidated Deposits of the Colville River Watershed, Stevens County, Washington. U.S. Geological Survey Water Resources Investigations Report 03-4128, 84 pp. Accessed March 15, 2021 from https://pubs.usgs.gov/wri/wri034128/pdf/wri034128.pdf	-	-	Approximated from Fig. 6 of Kaule et al. (2003) with guidance from well completion report data. Upper and Lower basins defined near the town of Addy (Washington).
Osoyoos Aquifer	Okanagan Basin	Rathfelder, K., Gregory, L. (2019). Groundwater quality assessment and proposed objectives for the Osoyoos Aquifer. Water Science Series: WSS2019-06, Province of British Columbia, Victoria, 88 pp. Accessed March 15, 2021 from https://a100.gov.bc.ca/pub/acat/documents/r57603/r1_1571784531661_1784376098.pdf	-	-	Approximated from Fig. 4 of Rathfelder and Gregory (2019) and well completion records for Washington and British Columbia
Oliver Sub-basin	Okanagan Basin	Neilson-Welch, L., Allen, D. (2007). Groundwater and hydrogeological conditions in the Okanagan Basin, British Columbia: a state-of-the-basin report. Final report prepared for Objective 1 of the Phase 2 Groundwater Supply and Demand Project, 165 pp. Accessed March 15, 2021 from https://www.obwb.ca/fileadmin/docs/water_supply_demand/water_supply_demand_final_report.pdf	-	-	Approximated from Fig. 12 of Neilson-Welch and Allen (2007) and well completion records for British Columbia
Kelowna Aquifer System	Okanagan Basin	Harrington, J. (2013). Hydrogeochemistry of the Kelowna aquifer system. University of British Columbia thesis, 52 pp. Accessed March 16, 2021 from https://www.obwb.ca/newsite/wp-content/uploads/kelowna_water_quality_harrington_honours_thesis.pdf	-	-	Approximated from Fig. 4 of Harrington (2013)
Pentteton and Shuttleworth Creek Aquifers	Okanagan Basin	Golder Associates and Summit Environmental Consultants Ltd. (2009). Phase 2 Okanagan Water Supply and Demand Project: Groundwater Objectives 2 and 3 Basin Study. Report to Okanagan Basin Water Board, 114 pp.	-	-	Approximated from description on pages 33 and 34 of report by Golder Associates and Summit Environmental Consultants Ltd. (2009) and Fig. 5

Aquifer System	Broader System	References			Delineation steps and sources
		Accessed March 16, 2021 from https://www.obwb.ca/wsd/about/project-reports			
Okanagan Lake Perimeter (west)	Okanagan Basin	Golder Associates and Summit Environmental Consultants Ltd. (2009). Phase 2 Okanagan Water Supply and Demand Project: Groundwater Objectives 2 and 3 Basin Study Report to Okanagan Basin Water Board, 114 pp. Accessed March 16, 2021 from https://www.obwb.ca/wsd/about/project-reports	-	-	Approximated from Fig. 7.1 of report by Golder Associates and Summit Environmental Consultants Ltd. (2009). Multiple lake perimeter aquifers shown in figure were stitched together and entitled "Lake Perimeter" aquifers
Okanagan Lake Perimeter (east)	Okanagan Basin	Golder Associates and Summit Environmental Consultants Ltd. (2009). Phase 2 Okanagan Water Supply and Demand Project: Groundwater Objectives 2 and 3 Basin Study Report to Okanagan Basin Water Board, 114 pp. Accessed March 16, 2021 from https://www.obwb.ca/wsd/about/project-reports	-	-	Approximated from Fig. 7.1 of report by Golder Associates and Summit Environmental Consultants Ltd. (2009). Multiple lake perimeter aquifers shown in figure were stitched together and entitled "Lake Perimeter" aquifers
Vernon Aquifer	Okanagan Basin	Golder Associates and Summit Environmental Consultants Ltd. (2009). Phase 2 Okanagan Water Supply and Demand Project: Groundwater Objectives 2 and 3 Basin Study Report to Okanagan Basin Water Board, 114 pp. Accessed March 16, 2021 from https://www.obwb.ca/wsd/about/project-reports	-	-	Approximated from description on page 40 of report by Golder Associates and Summit Environmental Consultants Ltd. (2009) and Fig. 5
Winfield-Wood Lake and Kalamalka Lake Aquifers	Okanagan Basin	Golder Associates and Summit Environmental Consultants Ltd. (2009). Phase 2 Okanagan Water Supply and Demand Project: Groundwater Objectives 2 and 3 Basin Study Report to Okanagan Basin Water Board, 114 pp. Accessed March 16, 2021 from https://www.obwb.ca/wsd/about/project-reports	-	-	Approximated from description on pages 37 and 38 of report by Golder Associates and Summit Environmental Consultants Ltd. (2009) and Fig. 5
North Okanagan	Okanagan Basin	Golder Associates and Summit Environmental Consultants Ltd. (2009). Phase 2 Okanagan Water Supply and Demand Project: Groundwater Objectives 2 and 3 Basin Study Report to Okanagan Basin Water Board, 114 pp. Accessed March 16, 2021 from https://www.obwb.ca/wsd/about/project-reports	-	-	Approximated from description on page 37 ("7.3.3. North Okanagan") of report by Golder Associates and Summit Environmental Consultants Ltd. (2009) and Fig. 5

Aquifer System	Broader System	References			Delineation steps and sources
Rogue Basin	-	Robison, J. H. (1974). Availability and quality of ground water in the Medford area, Jackson County, Oregon. U.S. Geological Survey Hydrologic Atlas 392, 2 plates. Accessed March 16, 2021 from https://pubs.er.usgs.gov/publication/ha392	State of Oregon Department of Environmental Quality (2015). Statewide Groundwater Monitoring Program: Mid-Rogue Basin 2015. 30 pp. Accessed March 16, 2021 from https://www.oregon.gov/deq/FilterDocs/gw-DEQ16-LAB-0042-TR.pdf	-	Approximated from Fig. 2 of State of Oregon Department of Environmental Quality (2015) and plate 1 of Robison (1974)
Southern Tule Lake	Upper Klamath Basin	Gannett, M.W., Breen, K.H. (2015). Groundwater levels, trends, and relations to pumping in the Bureau of Reclamation Klamath Project, Oregon and California: U.S. Geological Survey Open-File Report 2015-1145, 19 pp. Accessed March 20, 2021 from https://pubs.usgs.gov/of/2015/1145/ofr20151145.pdf	-	-	Approximated from Fig. 1 of Gannett and Breen (2015).
Northern Tule Lake	Upper Klamath Basin	Gannett, M.W., Breen, K.H. (2015). Groundwater levels, trends, and relations to pumping in the Bureau of Reclamation Klamath Project, Oregon and California: U.S. Geological Survey Open-File Report 2015-1145, 19 pp. Accessed March 20, 2021 from https://pubs.usgs.gov/of/2015/1145/ofr20151145.pdf	-	-	Approximated from Fig. 1 of Gannett and Breen (2015).
Lower Klamath Lake	Upper Klamath Basin	Gannett, M.W., Breen, K.H. (2015). Groundwater levels, trends, and relations to pumping in the Bureau of Reclamation Klamath Project, Oregon and California: U.S. Geological Survey Open-File Report 2015-1145, 19 pp. Accessed March 20, 2021 from https://pubs.usgs.gov/of/2015/1145/ofr20151145.pdf	-	-	Approximated from Fig. 1 of Gannett and Breen (2015).
Klamath River	Upper Klamath Basin	Gannett, M.W., Breen, K.H. (2015). Groundwater levels, trends, and relations to pumping in the Bureau of Reclamation Klamath Project, Oregon and California: U.S. Geological Survey Open-File Report 2015-1145, 19 pp. Accessed March 20, 2021 from https://pubs.usgs.gov/of/2015/1145/ofr20151145.pdf	-	-	Approximated from Fig. 1 of Gannett and Breen (2015).
Klamath Valley	Upper Klamath Basin	Gannett, M.W., Breen, K.H. (2015). Groundwater levels, trends, and relations to pumping in the Bureau of Reclamation Klamath Project, Oregon and California: U.S. Geological Survey Open-File Report	-	-	Approximated from Fig. 1 of Gannett and Breen (2015).

Aquifer System	Broader System	References			Delineation steps and sources
		2015-1145, 19 pp. Accessed March 20, 2021 from https://pubs.usgs.gov/of/2015/1145/ofr20151145.pdf			
Upper Lost River	Upper Klamath Basin	Gannett, M.W., Breen, K.H. (2015). Groundwater levels, trends, and relations to pumping in the Bureau of Reclamation Klamath Project, Oregon and California: U.S. Geological Survey Open-File Report 2015-1145, 19 pp. Accessed March 20, 2021 from https://pubs.usgs.gov/of/2015/1145/ofr20151145.pdf	-	-	Approximated from Fig. 4 of Gannett and Breen (2015).
Butte Valley	Upper Klamath Basin	Wood, P. R. (1960). Geology and ground-water features of the Butte Valley region, Siskiyou County, California. Geological Survey Water Supply Paper 1491, 155 pp. Accessed March 20, 2021 from https://pdfs.semanticscholar.org/a448/a58a3e1ae120d440d26f75b74512f0a868a4.pdf	-	-	Approximated from Fig. 1 of Wood (1960)
Sprague Basin	Upper Klamath Basin	Gannett, M. W., Lite, K. E., La Marche, J. L., Fisher, B. J., Polette, D. J. (2007). Ground-water hydrology of the upper Klamath Basin, Oregon and California. U. S. Geological Survey Scientific Investigations Report 2007-5050, 98 pp. Accessed March 20, 2021 from https://pubs.usgs.gov/sir/2007/5050/pdf/sir20075050.pdf	-	-	Approximated from Fig. 18 of
Swan Lake and Upper Klamath Lake Perimeter	Upper Klamath Basin	Gannett, M. W., Lite, K. E., La Marche, J. L., Fisher, B. J., Polette, D. J. (2007). Ground-water hydrology of the upper Klamath Basin, Oregon and California. U. S. Geological Survey Scientific Investigations Report 2007-5050, 98 pp. Accessed March 20, 2021 from https://pubs.usgs.gov/sir/2007/5050/pdf/sir20075050.pdf	-	-	Approximated from Fig. 18 of
Shasta Valley	-	Wood, P. R. (1960). Geology and ground-water features of the Butte Valley region, Siskiyou County, California. Geological Survey Water Supply Paper 1491, 155 pp. Accessed March 20, 2021 from https://pdfs.semanticscholar.org/a448/a5	-	-	Approximated from Fig. 1 of Wood (1960)

Aquifer System	Broader System	References	Delineation steps and sources
		8a3e1ac120d400d26f75b74512f0a868a4.pdf	
Scott Valley	-	Wood, P. R. (1960). Geology and ground-water features of the Butte Valley region, Siskiyou County, California. Geological Survey Water Supply Paper 1401, 155 pp. Accessed March 20, 2021 from https://pdfs.semanticscholar.org/a448/a58a3e1ac120d400d26f75b74512f0a868a4.pdf	Approximated from Fig. 1 of Wood (1960)
Eureka and Eel River and Mad River Plains	-	Johnson, M. J. (1975). Ground-water conditions in the Eureka Area, Humboldt County, California. U.S. Geological Survey Water Resources Investigations 78-127, 61 pp. Accessed March 20, 2021 from https://pubs.usgs.gov/wri/1978/0127/report.pdf	Approximated from Fig. 1 of Johnson (1975)
Goleta Basin	-	Bachman, S. (2010). Goleta Groundwater Basin Groundwater Management Plan. 91 pp. Accessed March 20, 2021 from https://www.goletawater.com/assets/uploads/documents/groundwater-management/Groundwater_Management_Plan_Final_05-11-10.pdf	Approximated from Fig. 1-4 of Bachman (2010)
Santa Barbara and Foothill Basin	-	Nishikawa, T., ed. (2018). Santa Barbara and Foothill groundwater basins: Geohydrology and optimal water resources management—Developed using density-dependent solute transport and optimization models. U.S. Geological Survey, Scientific Investigations Report 2018-5059. Accessed March 20, 2021 from https://pubs.usgs.gov/sir/2018/5059/sir20185059.pdf	Approximated from Fig. 1 of Nishikawa (2018)
Montecito Basin	-	Muir, K. S. (1968). Ground-water reconnaissance of the Santa Barbara-Montecito area, Santa Barbara County, California. U.S. Geological Survey Water Supply Paper 1859-A, 28 pp. Accessed March 20, 2021 from https://pubs.usgs.gov/wsp/1859a/report.pdf	Approximated from Plate 1 of Muir (1968)
Carpenteria Basin	-	La Rocque, G.A., Upson, J. E., Werts Jr., G. F. (1950). Wells and water levels in	Approximated from Plate 1 of La Rocque et al. (1968)

Aquifer System	Broader System	References	Delineation steps and sources
		principal ground-water basins in Santa Barbara County, California—Part 1: Carpinteria, Goleta and Santa Ynez Valleys, 1930-41. U.S. Geological Survey Water Supply Paper 1068—465 pp. Accessed March 20, 2021 from https://pubs.usgs.gov/wsp/1068/report.pdf	
Santa Clara-Calleguas Basin	-	Hanson, R.T., Martin, P., Koczol, K.M. (2002). Simulation of ground-water/surface-water flow in the Santa Clara-Calleguas ground-water basin, Ventura County, California. Water Resources Investigations Report 2002-4136, 172 pp. Accessed March 20, 2021 from https://pubs.usgs.gov/wri/wri024136/wri024136.pdf	Approximated from Fig. 7 of Hanson et al. (2002)
Antelope Valley	-	Stamos, C.L., Christensen, A. H., Langenheim, V. (2017). Preliminary hydrogeologic assessment near the boundary of the Antelope Valley and El Mirage Valley groundwater basins, California. U.S. Geological Survey Scientific Investigations Report 2017-5065, 56 pp. Accessed March 20, 2021 from https://pubs.usgs.gov/sir/2017/5065/sir20175065.pdf	Approximated from Fig. 7 of Stamos et al. (2017)
Mojave Basin	-	Stamos, C.L., Christensen, A. H., Langenheim, V. (2017). Preliminary hydrogeologic assessment near the boundary of the Antelope Valley and El Mirage Valley groundwater basins, California. U.S. Geological Survey Scientific Investigations Report 2017-5065, 56 pp. Accessed March 20, 2021 from https://pubs.usgs.gov/sir/2017/5065/sir20175065.pdf	Approximated from Fig. 7 of Stamos et al. (2017)
Upper Santa Ana Basin	-	Kent, R., Belitz, K. (2009). Ground-water quality data in the Upper Santa Ana Watershed Study Unit, November 2006 to March 2007: Results from the California GAMA Program. U.S. Geological Survey Data Series 404, 116 pp. Accessed March 21, 2021 from https://pubs.usgs.gov/ds/404/ds404.pdf	Approximated from Fig. 2 of Kent and Belitz (2009)

Aquifer System	Broader System	References			Delineation steps and sources
San Jacinto Basin	-	Kent, R., Belitz, K. (2009). Ground-water quality data in the Upper Santa Ana Watershed Study Unit, November 2006 to March 2007: Results from the California-CAMA Program. U.S. Geological Survey Data Series 404, 116 pp. Accessed March 21, 2021 from https://pubs.usgs.gov/ds/404/ds404.pdf	-	-	Approximated from Fig. 2 of Kent and Belitz (2009)
Tijuana-San Diego	-	International Hydrological Programme, Division of Water Sciences (2009). Atlas of Transboundary Aquifers. UNESCO Report, 322 pp. Accessed March 21, 2021 from https://isam.org/sites/default/files/resources/files/2%20Atlas%20of%20TBA.pdf	-	-	Approximated from map on page 99 of International Hydrological Programme, Division of Water Sciences (2009)
Smith Valley	-	Loeltz, O. J., Eakin, T. E. (1953). Geology and water resources of Smith Valley, Lyon and Douglas Counties, Nevada. U.S. Geological Survey Water-Supply Paper 1228, 94 pp. Accessed March 21, 2021 from https://pubs.usgs.gov/wsp/1228/report.pdf	-	-	Approximated from Plate 2 of Loeltz and Eakin (1953)
Mason Valley	-	Carroll, R.W., Pohl, G., McGraw, D., Garner, C., Knust, A., Boyle, D., Miner, T., Bassett, S., Pohlmann, K. (2010). Mason Valley Groundwater Model: Linking Surface Water and Groundwater in the Walker River Basin, Nevada 1. JAWRA Journal of the American Water Resources Association, 46(3), 554-573.	-	-	Approximated from Fig. 2 of Carroll et al. (2010); southern margins extended to include recorded drilled wells.
Lower Walker Basin	-	Allander, K. K., Niewonger, R. G., Jeton, A. E. (2014). Simulation of the Lower Walker River Basin hydrologic system, west-central Nevada, using PRMS and MODFLOW models. U.S. Geological Survey Scientific Investigations Report 2014-5190, 108 pp. Accessed March 21, 2021 from https://pubs.usgs.gov/sir/2014/5190/pdf/#2014-5190.pdf	-	-	Approximated from Fig. 4 of Allander et al. (2014) with guidance from completed well locations
Sierra Valley	-	Bachand, P.A.M., Birt, K.S., Bachand, S.M. (2020). Groundwater relationships to pumping, precipitation and geology in high elevation basin, Sierra Valley, CA. Technical Report to Report to Feather River Land, Sierra Valley, CA. 66 pp. Accessed March 21, 2021 from	-	-	Approximated from Fig. 4 of Bachand et al. (2020)

Aquifer System	Broader System	References	Delineation steps and sources
Honey Lake Valley	-	http://aquaticcommons.org/27004/1/Sierra%20Valley%20Recharge%20FINAL%202020-03-10%20SECURE.pdf Handman, E. H., Londquist, C. J., Maurer, D.K. (1990). Ground-water resources of Honey Lake valley, Lassen County, California, and Washoe County, Nevada. Water Resources Investigations Report 90-4050, 119 pp. Accessed March 21, 2021 from https://pubs.usgs.gov/wri/1990/4050/report.pdf	Approximated from Fig. 1 of Handman et al. (1990)
Spanish Springs Valley	-	Berger, D. L., Ross, W. C., Thodal, C. E., Robledo, A. R. (1997). Hydrogeology and simulated effects of urban development on water resources of Spanish Springs Valley, Washoe County, West-Central Nevada. U.S. Geological Survey Water Resources Investigations Report 96-4207, 87 pp. Accessed March 22, 2021 from https://pubs.usgs.gov/wri/1996/4207/report.pdf	Approximated from Fig. 4 of Berger et al. (1997)
Warm Springs Valley	-	Glenn, R.J. (1968). Water resources of Warm Springs Valley, Washoe County, Nevada. University of Nevada, Reno, Geological Sciences and Engineering Thesis 98 pp. Accessed March 21, 2021 from https://scholarworks.unr.edu/handle/11714/4345	Approximated from Plate 1 of Glenn (1968).
Hungry Valley	-	Kinder, J. (2012). Development of a Groundwater Flow Model for Hungry Valley, Washoe County, Nevada. University of Nevada, Reno. MSc Thesis, 223 pp. Accessed March 22, 2021 from https://scholarworks.unr.edu/bitstream/handle/11714/3578/Kinder_unr_0139M_10957.pdf?sequence=1&isAllowed=y	Approximated from Fig. 6 of
Grass Valley in northern Nevada	-	Welch, A. H., Soroy, M. L., Olmsted, F. H. (1981). Hydrothermal system in Southern Grass Valley, Pershing County, Nevada. U.S. Geological Survey Open-File Report 81-015, 200 pp. Accessed March 22, 2021 from https://www.osti.gov/servlets/purl/5119283-5mJ8YB/	Approximated from Fig. 2 of Welch et al. (1981)

Aquifer System	Broader System	References			Delineation steps and sources
Buena Vista Valley in northern Nevada	-	Welch, A. H., Sorey, M. L., Olmsted, F. H. (1981). Hydrothermal system in Southern Grass Valley, Pershing County, Nevada. U.S. Geological Survey Open-File Report 81-015, 200 pp. Accessed March 22, 2021 from https://www.osti.gov/servlets/purl/5119283-5mJ8YB/	-	-	Approximated from Fig. 2 of Welch et al. (1981)
Buffalo Valley in northern Nevada	-	Welch, A. H., Sorey, M. L., Olmsted, F. H. (1981). Hydrothermal system in Southern Grass Valley, Pershing County, Nevada. U.S. Geological Survey Open-File Report 81-015, 200 pp. Accessed March 22, 2021 from https://www.osti.gov/servlets/purl/5119283-5mJ8YB/	-	-	Approximated from Fig. 2 of Welch et al. (1981)
Lower Reese River Valley and Antelope Valley	-	Bredehoeft, J.D., Farvolden, R.N. (1963). International Association of Scientific Hydrology, Commission of Subterranean Waters, Publication no. 64, p. 197–212. Accessed March 23, 2021 from http://hydrologie.org/redbooks/a064/064047.pdf	-	-	Approximated from Fig. 1 of Bredehoeft and Farvolden (1963) and guided by locations of wells and topography (especially to distinguish upper and lower portions of basin).
Crescent Valley in northern Nevada	-	Bredehoeft, J.D., Farvolden, R.N. (1963). International Association of Scientific Hydrology, Commission of Subterranean Waters, Publication no. 64, p. 197–212. Accessed March 23, 2021 from http://hydrologie.org/redbooks/a064/064047.pdf	-	-	Approximated from Fig. 1 of Bredehoeft and Farvolden (1963)
Diamond Valley in northern Nevada	-	Bredehoeft, J.D., Farvolden, R.N. (1963). International Association of Scientific Hydrology, Commission of Subterranean Waters, Publication no. 64, p. 197–212. Accessed March 23, 2021 from http://hydrologie.org/redbooks/a064/064047.pdf	-	-	Approximated from Fig. 1 of Bredehoeft and Farvolden (1963)
Boulder Valley in northern Nevada	-	Bredehoeft, J.D., Farvolden, R.N. (1963). International Association of Scientific Hydrology, Commission of Subterranean Waters, Publication no. 64, p. 197–212. Accessed March 23, 2021 from http://hydrologie.org/redbooks/a064/064047.pdf	-	-	Approximated from Fig. 1 of Bredehoeft and Farvolden (1963)
Paradise Valley in northern Nevada	-	Bredehoeft, J.D., Farvolden, R.N. (1963). International Association of Scientific Hydrology, Commission of Subterranean Waters, Publication no. 64, p. 197–212.	-	-	Approximated from Fig. 1 of Bredehoeft and Farvolden (1963)

Aquifer System	Broader System	References	Delineation steps and sources
		Accessed March 23, 2021 from http://hydrologie.org/redbooks/a064/064047.pdf	
Kings River Valley in northern Nevada	-	Bredehoeft, J.D., Farvolden, R.N. (1963). International Association of Scientific Hydrology, Commission of Subterranean Waters, Publication no. 64, p. 197–212. Accessed March 23, 2021 from http://hydrologie.org/redbooks/a064/064047.pdf	Approximated from Fig. 1 of Bredehoeft and Farvolden (1963)
Silver State Valley in northern Nevada	-	Lopes, T. J. (2010). Hydrologic Evaluation of the Jungo Area, Southern Desert Valley, Nevada. U. S. Geological Survey Open-File Report 2010-1009, 18 pp. Accessed March 23, 2021 from https://pubs.usgs.gov/of/2010/1009/pdf/ofr20101009.pdf	Fig. 2 of Lopes (2010)
Desert Valley in northern Nevada	-	Lopes, T. J. (2010). Hydrologic Evaluation of the Jungo Area, Southern Desert Valley, Nevada. U. S. Geological Survey Open-File Report 2010-1009, 18 pp. Accessed March 23, 2021 from https://pubs.usgs.gov/of/2010/1009/pdf/ofr20101009.pdf	Fig. 2 of Lopes (2010)
Quinn River Valley in northern Nevada	-	Huntington, J.L., Minor, B., Bromley, M., Morton, C. (2018). Reconnaissance Investigation of Phreatophyte Vegetation Vigor for Selected Hydrographic Areas in Nevada. Division of Hydrologic Sciences, Desert Research Institute, Reno, NV. Report, 75 pp. Accessed March 23, 2021 from http://www.conservationgateway.org/ConservationByGeography/NorthAmerica/UnitedStates/nevada/water/Documents/Final%20DRI-TNC%20spatiotemporal%20phreatophyte%20report_may31.pdf	Fig. 6 of Huntington et al. (2018); boundary extended northward and delineated based on well completion reports.
Upper Humboldt Basin	-	Plume, R. W. (2009). Hydrogeologic Framework and Occurrence and Movement of Ground Water in the Upper Humboldt River Basin, Northeastern Nevada. U.S. Geological Survey Scientific Investigations Report 2009-5014, 30 pp. Accessed March 2021 from https://pubs.usgs.gov/sir/2009/5014/pdf/sir20095014.pdf	Approximated from Plate 1 of Plume (2009)

Aquifer System	Broader System	References			Delineation steps and sources
Ruby Valley	-	Berger, D. L. (2006). Hydrogeology and Water Resources of Ruby Valley, Northeastern Nevada. Scientific Investigations Report 2005-5247, 48 pp. Accessed March 23, 2021 from https://pubs.usgs.gov/sir/2005/5247/sir2005-5247.pdf	-	-	Approximated from Fig. 3 of Berger (2006).
Clover Valley	-	Berger, D. L. (2006). Hydrogeology and Water Resources of Ruby Valley, Northeastern Nevada. Scientific Investigations Report 2005-5247, 48 pp. Accessed March 23, 2021 from https://pubs.usgs.gov/sir/2005/5247/sir2005-5247.pdf	-	-	Approximated from Fig. 1 of Berger (2006).
Fish Lake Valley	-	Reheis, M.C., Block, D. (2007). Surficial geologic map and geochronologic database, Fish Lake Valley, Esmeralda County, Nevada, and Mono County, California, U.S. Geological Survey Data Series 277, 14 pp. Accessed March 23, 2021 from https://pubs.usgs.gov/ds/277/report.pdf	-	-	Approximated from Fig. 4 of Reheis and Block (2007)
Kootenai Valley	-	Graham, W. G., Campbell, L. J. (1981). Groundwater resources of Idaho. Idaho Department of Water Resources Report, 61 pp. Accessed March 23, 2021 from https://idwr.idaho.gov/files/publications/498408-MISC-GW-Resources-ID.pdf	-	-	Approximated from Plate 1 of Graham and Campbell (1981). Margins extended into Montana and British Columbia, guided by topography and well completion report data.
Priest River Basin	-	Graham, W. G., Campbell, L. J. (1981). Groundwater resources of Idaho. Idaho Department of Water Resources Report, 61 pp. Accessed March 23, 2021 from https://idwr.idaho.gov/files/publications/498408-MISC-GW-Resources-ID.pdf	-	-	Approximated from Plate 1 of Graham and Campbell (1981). Southern boundary of basin delineated based on topography.
Pend Orielle West	-	Graham, W. G., Campbell, L. J. (1981). Groundwater resources of Idaho. Idaho Department of Water Resources Report, 61 pp. Accessed March 23, 2021 from https://idwr.idaho.gov/files/publications/498408-MISC-GW-Resources-ID.pdf	-	-	Approximated from Plate 1 of Graham and Campbell (1981). Western portion of Pend Orielle River aquifer was delineated separately from the eastern portion.
Pend Orielle East	-	Graham, W. G., Campbell, L. J. (1981). Groundwater resources of Idaho. Idaho Department of Water Resources Report, 61 pp. Accessed March 23, 2021 from https://idwr.idaho.gov/files/publications/498408-MISC-GW-Resources-ID.pdf	-	-	Approximated from Plate 1 of Graham and Campbell (1981). Eastern portion of Pend Orielle River aquifer was delineated separately from the western portion.

Aquifer System	Broader System	References			Delineation steps and sources
Mill Creek Aquifer	-	Graham, W. G., Campbell, L. J. (1981). Groundwater resources of Idaho. Idaho Department of Water Resources Report, 61 pp. Accessed March 23, 2021 from https://idwr.idaho.gov/files/publications/498108-MISC-GW-Resources-ID.pdf	-	-	Approximated from Plate 1 of Graham and Campbell (1981).
Long Valley—Round Valley	-	Graham, W. G., Campbell, L. J. (1981). Groundwater resources of Idaho. Idaho Department of Water Resources Report, 61 pp. Accessed March 23, 2021 from https://idwr.idaho.gov/files/publications/498108-MISC-GW-Resources-ID.pdf	-	-	Approximated from Plate 1 of Graham and Campbell (1981).
Weiser River Valley	-	Graham, W. G., Campbell, L. J. (1981). Groundwater resources of Idaho. Idaho Department of Water Resources Report, 61 pp. Accessed March 23, 2021 from https://idwr.idaho.gov/files/publications/498108-MISC-GW-Resources-ID.pdf	-	-	Approximated from Plate 1 of Graham and Campbell (1981).
Payette Valley	-	Graham, W. G., Campbell, L. J. (1981). Groundwater resources of Idaho. Idaho Department of Water Resources Report, 61 pp. Accessed March 23, 2021 from https://idwr.idaho.gov/files/publications/498108-MISC-GW-Resources-ID.pdf	-	-	Approximated from Plate 1 of Graham and Campbell (1981). Oregon portion based on well completion data.
Scott Valley and Mann Creek	-	Graham, W. G., Campbell, L. J. (1981). Groundwater resources of Idaho. Idaho Department of Water Resources Report, 61 pp. Accessed March 23, 2021 from https://idwr.idaho.gov/files/publications/498108-MISC-GW-Resources-ID.pdf	-	-	Approximated from Plate 1 of Graham and Campbell (1981).
Boise Valley and Homedale-Murphy Area	Western Snake River Plain	Graham, W. G., Campbell, L. J. (1981). Groundwater resources of Idaho. Idaho Department of Water Resources Report, 61 pp. Accessed March 23, 2021 from https://idwr.idaho.gov/files/publications/498108-MISC-GW-Resources-ID.pdf	Lindholm, G. F. (1996). Summary of the Snake River Plain regional aquifer-system analysis in Idaho and eastern Oregon. U.S. Geological Survey Professional Paper 1408-A, 69 pp. Accessed March 1, 2021 via https://pubs.usgs.gov/pp/1408a/report.pdf	-	Approximated from Plate 1 of Graham and Campbell (1981). Zones "30" and "32" merged into one; locations of completed wells were used to guide the western boundary of the area. Broader portions of the Snake River Plain (i.e., Central, Western and Eastern) were approximated from Fig. 4 of Lindholm (1996).
Mountain Home Plateau	Western Snake River Plain	Graham, W. G., Campbell, L. J. (1981). Groundwater resources of Idaho. Idaho Department of Water Resources Report, 61 pp. Accessed March 23, 2021 from https://idwr.idaho.gov/files/publications/498108-MISC-GW-Resources-ID.pdf	Young, H.W. (1978). Reconnaissance of ground-water resources in the Mountain Home plateau area, southwest Idaho. U.S. Geological Survey Water Resources Investigations Report 77-108, 48 pp. Accessed November 29, 2021 from	Lindholm, G. F. (1996). Summary of the Snake River Plain regional aquifer-system analysis in Idaho and eastern Oregon. U.S. Geological Survey Professional Paper 1408-A, 69 pp. Accessed March 1, 2021 via	Approximated from Plate 1 of Graham and Campbell (1981) and Fig. 1 by Young (1981). Broader portions of the Snake River Plain (i.e., Central, Western and Eastern) were approximated from Fig. 4 of Lindholm (1996).

Aquifer System	Broader System	References			Delineation steps and sources
			https://pubs.usgs.gov/wri/1977/0408/report.pdf	https://pubs.usgs.gov/pp/1408a/report.pdf	
Bruneau-Grandview Area	Western Snake River Plain	Graham, W. G., Campbell, L. J. (1981). Groundwater resources of Idaho. Idaho Department of Water Resources Report, 64 pp. Accessed March 23, 2021 from https://dwr.idaho.gov/files/publications/498108-MISC-GW-Resources-ID.pdf	Lindholm, G. F. (1996). Summary of the Snake River Plain regional aquifer system analysis in Idaho and eastern Oregon. U.S. Geological Survey Professional Paper 1408-A, 69 pp. Accessed March 1, 2021 via https://pubs.usgs.gov/pp/1408a/report.pdf	-	Approximated from Plate 1 of Graham and Campbell (1981). Broader portions of the Snake River Plain (i.e., Central, Western and Eastern) were approximated from Fig. 1 of Lindholm (1996).
Central Plain	Central Snake River Plain	Graham, W. G., Campbell, L. J. (1981). Groundwater resources of Idaho. Idaho Department of Water Resources Report, 64 pp. Accessed March 23, 2021 from https://dwr.idaho.gov/files/publications/498108-MISC-GW-Resources-ID.pdf	Lindholm, G. F. (1996). Summary of the Snake River Plain regional aquifer system analysis in Idaho and eastern Oregon. U.S. Geological Survey Professional Paper 1408-A, 69 pp. Accessed March 1, 2021 via https://pubs.usgs.gov/pp/1408a/report.pdf	-	Approximated from Plate 1 of Graham and Campbell (1981). Broader portions of the Snake River Plain (i.e., Central, Western and Eastern) were approximated from Fig. 1 of Lindholm (1996); "Eastern" portion as defined by Lindholm (1996) was split into two portions (entitled "Central" and "Eastern" Snake River Plain, respectively) at Craters of the Moon National Monument.
Twin Falls	Central Snake River Plain	Graham, W. G., Campbell, L. J. (1981). Groundwater resources of Idaho. Idaho Department of Water Resources Report, 64 pp. Accessed March 23, 2021 from https://dwr.idaho.gov/files/publications/498108-MISC-GW-Resources-ID.pdf	-	-	Approximated from Plate 1 of Graham and Campbell (1981).
Marsh Valley	Central Snake River Plain	Graham, W. G., Campbell, L. J. (1981). Groundwater resources of Idaho. Idaho Department of Water Resources Report, 64 pp. Accessed March 23, 2021 from https://dwr.idaho.gov/files/publications/498108-MISC-GW-Resources-ID.pdf	-	-	Approximated from Plate 1 of Graham and Campbell (1981).
Goose Creek and Golden Valley	Central Snake River Plain	Graham, W. G., Campbell, L. J. (1981). Groundwater resources of Idaho. Idaho Department of Water Resources Report, 64 pp. Accessed March 23, 2021 from https://dwr.idaho.gov/files/publications/498108-MISC-GW-Resources-ID.pdf	-	-	Approximated from Plate 1 of Graham and Campbell (1981).
Raft River Valley	Central Snake River Plain	Graham, W. G., Campbell, L. J. (1981). Groundwater resources of Idaho. Idaho Department of Water Resources Report, 64 pp. Accessed March 23, 2021 from https://dwr.idaho.gov/files/publications/498108-MISC-GW-Resources-ID.pdf	-	-	Approximated from Plate 1 of Graham and Campbell (1981).
Marsh Creek	Eastern Snake River Plain	Graham, W. G., Campbell, L. J. (1981). Groundwater resources of Idaho. Idaho	-	-	Approximated from Plate 1 of Graham and Campbell (1981).

Aquifer System	Broader System	References			Delineation steps and sources
		Department of Water Resources Report, 64 pp. Accessed March 23, 2021 from https://idwr.idaho.gov/files/publications/498108-MISC-GW-Resources-ID.pdf			
Arbon Valley	Eastern Snake River Plain	Graham, W. G., Campbell, L. J. (1981). Groundwater resources of Idaho. Idaho Department of Water Resources Report, 64 pp. Accessed March 23, 2021 from https://idwr.idaho.gov/files/publications/498108-MISC-GW-Resources-ID.pdf	-	-	Approximated from Plate 1 of Graham and Campbell (1981).
Rockland Valley	Eastern Snake River Plain	Graham, W. G., Campbell, L. J. (1981). Groundwater resources of Idaho. Idaho Department of Water Resources Report, 64 pp. Accessed March 23, 2021 from https://idwr.idaho.gov/files/publications/498108-MISC-GW-Resources-ID.pdf	-	-	Approximated from Plate 1 of Graham and Campbell (1981).
Eastern Plain	Eastern Snake River Plain	Graham, W. G., Campbell, L. J. (1981). Groundwater resources of Idaho. Idaho Department of Water Resources Report, 64 pp. Accessed March 23, 2021 from https://idwr.idaho.gov/files/publications/498108-MISC-GW-Resources-ID.pdf	-	-	Approximated from Plate 1 of Graham and Campbell (1981) (see #39 "Snake Plain"; Volcanic Rift Zone" area (see preceding aquifer in this list) excluded from delineation)
Little Lost River Valley	Eastern Snake River Plain	Graham, W. G., Campbell, L. J. (1981). Groundwater resources of Idaho. Idaho Department of Water Resources Report, 64 pp. Accessed March 23, 2021 from https://idwr.idaho.gov/files/publications/498108-MISC-GW-Resources-ID.pdf	-	-	Approximated from Plate 1 of Graham and Campbell (1981).
Big Lost River Valley	Eastern Snake River Plain	Graham, W. G., Campbell, L. J. (1981). Groundwater resources of Idaho. Idaho Department of Water Resources Report, 64 pp. Accessed March 23, 2021 from https://idwr.idaho.gov/files/publications/498108-MISC-GW-Resources-ID.pdf	-	-	Approximated from Plate 1 of Graham and Campbell (1981).
Mud Lake	Eastern Snake River Plain	Rattray, G. (2015). Geochemical evolution of groundwater in the Mud Lake area, Eastern Idaho, USA. Environmental Earth Sciences, 73(12), 8251-8269.	-	-	Approximated from Fig. 2 of Rattray (2015)
Volcanic Rift Zone	Eastern Snake River Plain	Whitehead, R. L. (1992). Geohydrologic framework of the Snake River Plain regional aquifer system, Idaho and eastern Oregon. US Geological Survey Professional Paper 1408-B, 39 pp. Accessed March 1, 2021 via https://pubs.usgs.gov/pp/1408b/report.pdf	-	-	Approximated from Plate 1 of Whitehead (1992).

Aquifer System	Broader System	References			Delineation steps and sources
Teton Valley	-	Graham, W. G., Campbell, L. J. (1981). Groundwater resources of Idaho. Idaho Department of Water Resources Report, 61 pp. Accessed March 23, 2021 from https://idwr.idaho.gov/files/publications/498108-MISC-GW-Resources-ID.pdf	-	-	Approximated from Plate 1 of Graham and Campbell (1981).
Camas Prairie	-	Graham, W. G., Campbell, L. J. (1981). Groundwater resources of Idaho. Idaho Department of Water Resources Report, 61 pp. Accessed March 23, 2021 from https://idwr.idaho.gov/files/publications/498108-MISC-GW-Resources-ID.pdf	-	-	Approximated from Plate 1 of Graham and Campbell (1981).
Wood River Valley	-	Graham, W. G., Campbell, L. J. (1981). Groundwater resources of Idaho. Idaho Department of Water Resources Report, 61 pp. Accessed March 23, 2021 from https://idwr.idaho.gov/files/publications/498108-MISC-GW-Resources-ID.pdf	-	-	Approximated from Plate 1 of Graham and Campbell (1981).
Sawtooth Valley and Bear Valley	-	Graham, W. G., Campbell, L. J. (1981). Groundwater resources of Idaho. Idaho Department of Water Resources Report, 61 pp. Accessed March 23, 2021 from https://idwr.idaho.gov/files/publications/498108-MISC-GW-Resources-ID.pdf	-	-	Approximated from Plate 1 of Graham and Campbell (1981).
Pahsimeroi Valley	-	Graham, W. G., Campbell, L. J. (1981). Groundwater resources of Idaho. Idaho Department of Water Resources Report, 61 pp. Accessed March 23, 2021 from https://idwr.idaho.gov/files/publications/498108-MISC-GW-Resources-ID.pdf	-	-	Approximated from Plate 1 of Graham and Campbell (1981).
Round Valley	-	Graham, W. G., Campbell, L. J. (1981). Groundwater resources of Idaho. Idaho Department of Water Resources Report, 61 pp. Accessed March 23, 2021 from https://idwr.idaho.gov/files/publications/498108-MISC-GW-Resources-ID.pdf	-	-	Approximated from Plate 1 of Graham and Campbell (1981).
Upper Salmon Basin	-	Graham, W. G., Campbell, L. J. (1981). Groundwater resources of Idaho. Idaho Department of Water Resources Report, 61 pp. Accessed March 23, 2021 from https://idwr.idaho.gov/files/publications/498108-MISC-GW-Resources-ID.pdf	-	-	Approximated from Plate 1 of Graham and Campbell (1981).
Lemhi Valley	-	Graham, W. G., Campbell, L. J. (1981). Groundwater resources of Idaho. Idaho Department of Water Resources Report, 61 pp. Accessed March 23, 2021 from https://idwr.idaho.gov/files/publications/498108-MISC-GW-Resources-ID.pdf	-	-	Approximated from Plate 1 of Graham and Campbell (1981).

Aquifer System	Broader System	References			Delineation steps and sources
		https://idwr.idaho.gov/files/publications/498108-MISC-GW-Resources-ID.pdf			
Garden Valley	-	Graham, W. G., Campbell, L. J. (1981). Groundwater resources of Idaho. Idaho Department of Water Resources Report, 64 pp. Accessed March 23, 2021 from https://idwr.idaho.gov/files/publications/498108-MISC-GW-Resources-ID.pdf	-	-	Approximated from Plate 4 of Graham and Campbell (1981).
Napa Valley	-	Kunkle, F., Upson, J. E. (1960). Geology and ground water in Napa and Sonoma Valleys, Napa and Sonoma Counties, California. U.S. Geological Survey Water Supply Paper 1495, 260 pp. Accessed March 27, 2021 from https://pubs.usgs.gov/wsp/1495/report.pdf	-	-	Approximated from Plate 4 of Kunkle and Upson (1960).
Santa Clara Valley	-	Hanson, R. T., (2015). Hydrologic framework of the Santa Clara Valley, California. Geosphere 11, doi:10.1130/GES01104-1 Accessed March 27, 2021 from https://ca.water.usgs.gov/pubs/2015/Hanson2015.pdf	-	-	Approximated from Fig. 4 of Hanson (2015).
Surprise Valley	-	Cantwell, C.A., Fowler, A.P.G. (2014). Fluid Geochemistry of the Surprise Valley Geothermal System. Proceedings of the Thirty-Ninth Workshop on Geothermal Reservoir Engineering Stanford University, Stanford, California, February 24-26, 2014 SCP-TR-202. Accessed March 27, 2021 from https://pangea.stanford.edu/ERE/pdf/IGAstandard/SCW/2014/Cantwell.pdf	-	-	Approximated from Fig. 4 of Cantwell and Fowler (2014).
Salt Lake Valley	-	Thiros, S.A. (2003). Hydrogeology of shallow basin fill deposits in areas of Salt Lake Valley, Salt Lake County, Utah. U.S. Geological Survey Water Resources Investigations Report 2003-4029, 32 pp. Accessed March 27, 2021 from https://pubs.usgs.gov/wri/wri034029/pdf/wri034029.pdf	-	-	Approximated from Fig. 4 of Thiros (2003).
Utah Lake Valley	-	Dustin, J.D. (1978). Hydrogeology of Utah Lake with Emphasis on Goshen Bay. PhD Dissertation, Brigham Young University, 184 pp. Accessed March 27, 2021 from	-	-	Approximated from Fig. 4 of Dustin (1978)

Aquifer System	Broader System	References			Delineation steps and sources
		https://apps.dtic.mil/sti/pdfs/ADA065478.pdf			
Ogden Valley	-	Jordan, J. L., Smith, S. D., Inkenbrandt, P. C., Lowe, M., Hardwick, C. L., Wallace, J., Kirby, S. M., King, J. K., Payne, E. E. (2019). Characterization of the groundwater system in Ogden Valley, Weber County, Utah, with emphasis on groundwater-surface-water interaction and the groundwater budget. Utah Geological Survey Report, Special Study 165. 237 pp. Accessed March 27, 2021 from https://ugspub.nr.utah.gov/publications/special_studies/ss-165/ss-165.pdf	-	-	Approximated from Fig. 1 of Dustin (1978)
East Shore Area	-	Price, D. (1985). Ground Water in Utah's Densely Populated Wasatch Front Area the Challenge and the Choices. U.S. Geological Survey Water Supply Paper 2232, 78 pp. Accessed March 27, 2021 from https://pubs.usgs.gov/wsp/2232/report.pdf	-	-	Approximated from Fig. 1 of Price (1985)
Bear River Bay	-	Price, D. (1985). Ground Water in Utah's Densely Populated Wasatch Front Area the Challenge and the Choices. U.S. Geological Survey Water Supply Paper 2232, 78 pp. Accessed March 27, 2021 from https://pubs.usgs.gov/wsp/2232/report.pdf	-	-	Approximated from Fig. 1 of Price (1985)
Cache Valley	-	Graham, W. G., Campbell, L. J. (1981). Groundwater resources of Idaho. Idaho Department of Water Resources Report, 61 pp. Accessed March 23, 2021 from https://dwr.idaho.gov/files/publications/198108-MISC-GW-Resources-ID.pdf	Bjorklund, L. J., McGreevy, L. J. (1971). Ground-water resources of Cache Valley, Utah and Idaho. Utah Department of Natural Resources, Division of Water Rights Technical Publication No. 36, 85 pp. Accessed March 27, 2021 from https://waterights.utah.gov/docSys/v920/w920/w920008y.pdf	-	Approximated from Plate 1 of Graham and Campbell (1981) and Fig. 1 of Bjorklund and McGreevy (1971).
Malad Valley	-	Graham, W. G., Campbell, L. J. (1981). Groundwater resources of Idaho. Idaho Department of Water Resources Report, 61 pp. Accessed March 23, 2021 from https://dwr.idaho.gov/files/publications/198108-MISC-GW-Resources-ID.pdf	-	-	Approximated from Plate 1 of Graham and Campbell (1981).
Bear Lake Valley	-	Graham, W. G., Campbell, L. J. (1981). Groundwater resources of Idaho. Idaho	-	-	Approximated from Plate 1 of Graham and Campbell (1981).

Aquifer System	Broader System	References	Delineation steps and sources
		Department of Water Resources Report, 64 pp. Accessed March 23, 2021 from https://idwr.idaho.gov/files/publications/198108-MISC-GW-Resources-ID.pdf	
Gem Valley	-	Graham, W. C., Campbell, L. J. (1981). Groundwater resources of Idaho. Idaho Department of Water Resources Report, 64 pp. Accessed March 23, 2021 from https://idwr.idaho.gov/files/publications/198108-MISC-GW-Resources-ID.pdf	-
Curlew Valley	-	Graham, W. C., Campbell, L. J. (1981). Groundwater resources of Idaho. Idaho Department of Water Resources Report, 64 pp. Accessed March 23, 2021 from https://idwr.idaho.gov/files/publications/198108-MISC-GW-Resources-ID.pdf	-
Duck Valley	-	Graham, W. C., Campbell, L. J. (1981). Groundwater resources of Idaho. Idaho Department of Water Resources Report, 64 pp. Accessed March 23, 2021 from https://idwr.idaho.gov/files/publications/198108-MISC-GW-Resources-ID.pdf	-
Heber Valley	-	Roark, D.M., Holmes, W.F., Shlosar, H.K. (1991). Hydrology of Heber and Round Valleys, Wasatch County, Utah, with emphasis on simulation of ground-water flow in Heber Valley. U.S. Geological Survey Technical Publication 101, 101 pp. Accessed March 27, 2021 from https://waterrights.utah.gov/docSys/v920/y920/y9200009.pdf	-
Mimbres Basin	-	Hanson, R.T., McLean, J.S., Miller, R.S. (1994). Hydrogeologic framework and preliminary simulation of ground-water flow in the Mimbres Basin, Southwestern New Mexico. U.S. Geological Survey, Water Resources Investigations Report 94-4011, 118 pp. Accessed March 27, 2021 from https://pubs.usgs.gov/wri/1994/4011/report.pdf	Finch, S. T., McCoy, A., Melis, E. (2008). Geologic Controls on Groundwater Flow in the Mimbres Basin, South Western New Mexico. In New Mexico Geological Society Guide Book, 59th Field Conference, pp. 189-198. Accessed March 27, 2021 from https://nmgs.nmt.edu/publications/guidebooks/downloads/59/59_p0189_p0198.pdf
		White, W. N. (1931). Preliminary report on the ground-water supply of Mimbres Valley, New Mexico. U.S. Geological Survey Water Supply Paper 637, 23 pp. Accessed November 7, 2021 from https://pubs.usgs.gov/wsp/0637B/report.pdf	-
Uvas Valley	-	State of New Mexico Office of the State Engineer (1998). Nutt Hockett Basin Hydrographic Survey Report. 267 pp. Accessed March 27, 2021 from	-
			Approximated from Plate 1 of Graham and Campbell (1981).
			Approximated from Plate 1 of Graham and Campbell (1981).
			Approximated from Plate 1 of Graham and Campbell (1981).
			Approximated from Fig. 1 of Roark et al. (1991)
			Approximated from Fig. 1 of Hansen et al. (1994), Fig. 1 of Finch et al. (2008) and Plate 1 by White (1931)
			Approximated from map on page vi of State of New Mexico, Office of the State Engineer (1998)

Aquifer System	Broader System	References	Delineation steps and sources
		https://www.ose.state.nm.us/HydroSurvey/nutt_bockett/report.pdf	
Central Hachita Valley	-	Trauger, F. D., Herrick, E. H. (1962). Ground Water in Central Hachita Valley Northeast of the Big Hatchet Mountains, Hidalgo County, New Mexico. New Mexico State Engineer Office Technical Report 26, 26 pp. Accessed March 27, 2021 from https://www.ose.state.nm.us/Library/TechnicalReports/TechReport-026.pdf	Approximated from map on page 3 of Trauger and Herrick (1961).
Wet Mountain Valley	-	Londquist, C. J., Livingston, R.K. (1978). Water resources appraisal of the Wet Mountain Valley, in parts of Custer and Fremont Counties, Colorado. Water-Resources Investigations-78-1, 62 pp. Accessed March 27, 2021 from https://pubs.usgs.gov/wri/1978/0004/report.pdf	Approximated from Fig. 3 of Londquist and Livingston (1978)
Upper Arkansas River Basin	-	Watts, K.R. (2005). Hydrogeology and quality of ground water in the upper Arkansas River Basin from Buena Vista to Salida, Colorado, 2000-2003. U.S. Geological Survey Scientific Investigations Report 2005-5179, 61 pp. Accessed March 27, 2021 from https://pubs.usgs.gov/sir/2005/5179/pdf/SIR2005-5179.pdf	Approximated from Fig. 4
Grand Valley	-	Lohman, S.W. (1965). Geology and Artesian Water Supply Grand Junction Area Colorado. U.S. Geological Survey Professional Paper 451, 157 pp. Accessed March 28, 2021 from https://pubs.usgs.gov/pp/0451/report.pdf	Approximated from Plate 1 of Lohman (1965)
Rogers Mesa	-	Watts, K.R. (2008). Availability, sustainability, and suitability of ground water, Rogers Mesa, Delta County, Colorado—types of analyses and data for use in subdivision water supply reports. U.S. Geological Survey Scientific Investigations Report 2008-5020, 53 pp. Accessed March 28, 2021 from https://pubs.usgs.gov/sir/2008/5020/pdf/SIR2008-5020.pdf	Approximated from Fig. 4 of Watts (2008).
Uncompahgre Basin	-	Craig, T.W. (1971). Ground water of the Uncompahgre Valley Montrose County, Colorado. MSc Thesis, University of Missouri-Rolla, 119 pp. Accessed March	Approximated from Fig. 5 of Craig et al. (1971)

Aquifer System	Broader System	References	Delineation steps and sources		
		28, 2021 from https://scholarsmine.mst.edu/cgi/viewcontent.cgi?article=6120&context=masters_theses			
Redlands Mesa	-	Kolm, K.E., van der Heijde, P.K.M. (2014). Groundwater Systems In Delta County, Colorado: Surface-Creek Valley Area. Report prepared for Delta County Board of County Commissioners, Colorado, 64 pp. Accessed March 28, 2021 from https://www.chc4you.org/wp-content/uploads/2017/01/Surface-Creek-Hydrology-Report-2014.pdf	Approximated from Fig. 23b of Kohl and van der Heijde (2014)		
Cedar Mesa	-	Kolm, K.E., van der Heijde, P.K.M. (2014). Groundwater Systems In Delta County, Colorado: Surface-Creek Valley Area. Report prepared for Delta County Board of County Commissioners, Colorado, 64 pp. Accessed March 28, 2021 from https://www.chc4you.org/wp-content/uploads/2017/01/Surface-Creek-Hydrology-Report-2014.pdf	Approximated from Fig. 23a of Kohl and van der Heijde (2014)		
Surface and Tongue Basins	-	Kolm, K.E., van der Heijde, P.K.M. (2014). Groundwater Systems In Delta County, Colorado: Surface-Creek Valley Area. Report prepared for Delta County Board of County Commissioners, Colorado, 64 pp. Accessed March 28, 2021 from https://www.chc4you.org/wp-content/uploads/2017/01/Surface-Creek-Hydrology-Report-2014.pdf	Approximated from Fig. 25b of Kohl and van der Heijde (2014)		
South Park Basin	-	Barkmann, P. E., Broes, L.D., Palkovic, M.J., Hopkins, J.C., Bird, K.S., Sebol, L.A., Fitzgerald, F.S. (2020). ON-010 Colorado Groundwater Atlas. Geohydrology. Colorado Geological Survey, Golden, CO. ON-010 Colorado Groundwater Atlas. Accessed March 29, 2021 from https://coloradogeologicalsurvey.org/water/colorado-groundwater-atlas/	Donagan, K.C. (2018). Groundwater levels in the South Park Basin 2018. Colorado Division of Water Resources Report, 29 pp. Accessed November 29, 2021 from https://dnrweblink.state.co.us/dwr/ElectronicFile.aspx?docid=3351305&bid=0	Barkmann, P.E., Moore, A., Johnson, J. (2013). South Park Groundwater Quality Scoping Study. Report for Coalition for the Upper South Platte, 74 pp. Accessed November 29, 2021 from https://cusp.ws/wp-content/uploads/2014/10/South-Park-Groundwater-Quality-Scoping-Study_Final.pdf	Approximated from Fig. 12-01 of Barkmann et al. (2020), Fig. 1a by Donagan (2018), and Fig. 1 by Barkmann et al. (2013)
Blue River Basin	-	Barkmann, P. E., Broes, L.D., Palkovic, M.J., Hopkins, J.C., Bird, K.S., Sebol, L.A., Fitzgerald, F.S. (2020). ON-010 Colorado Groundwater Atlas. Geohydrology. Colorado Geological Survey, Golden, CO. ON-010 Colorado	Approximated from Fig. 12-01 of Barkmann et al. (2020).		

Aquifer System	Broader System	References	Delineation steps and sources
		Groundwater Atlas. Accessed March 29, 2021 from https://coloradogeologicalsurvey.org/water/colorado-groundwater-atlas/	
North Park Basin	-	Robson, S.G., Graham, C. (1996). Geohydrology of the North Park area, Jackson County, Colorado, with a section on water law. U.S. Geological Survey Water Resources Investigations Report 96-4166, 1 plate. Accessed March 29, 2021 from https://pubs.usgs.gov/wri/1996/4166/plate-1.pdf	Approximated from plate 1 by Robson and Graham (1996).
Agency Park	-	Van Liew, W.P., Gesink, M.L. (1985). Preliminary assessment of the groundwater resources of the alluvial aquifer, White River valley, Rio Blanco County, Colorado. U.S. Geological Survey Water Resources Investigations Report 84-4307, 82 pp. Accessed March 29, 2021 from https://pubs.usgs.gov/wri/1984/4307/report.pdf	Approximated from plate 1 by Van Liew and Gesink (1985).
Powell Park	-	Van Liew, W.P., Gesink, M.L. (1985). Preliminary assessment of the groundwater resources of the alluvial aquifer, White River valley, Rio Blanco County, Colorado. U.S. Geological Survey Water Resources Investigations Report 84-4307, 82 pp. Accessed March 29, 2021 from https://pubs.usgs.gov/wri/1984/4307/report.pdf	Approximated from plate 1 by Van Liew and Gesink (1985).
Coal Oil Basin	-	Van Liew, W.P., Gesink, M.L. (1985). Preliminary assessment of the groundwater resources of the alluvial aquifer, White River valley, Rio Blanco County, Colorado. U.S. Geological Survey Water Resources Investigations Report 84-4307, 82 pp. Accessed March 29, 2021 from https://pubs.usgs.gov/wri/1984/4307/report.pdf	Approximated from plate 1 by Van Liew and Gesink (1985).
South Platte Basin	-	Wellman, T.P. (2015). Evaluation of groundwater levels in the South Platte River alluvial aquifer, Colorado, 1953–2012, and design of initial well networks for monitoring groundwater levels. U.S.	Approximated from Fig. 3 of Wellman (2015)

Aquifer System	Broader System	References	Delineation steps and sources
		Geological Survey Scientific Investigations Report 2015-5015, 67 pp., Accessed March 29, 2021 from https://pubs.usgs.gov/sir/2015/5015/pdf/sir2015-5015.pdf	
Lower Arkansas River Western Reach	-	Holmberg, M.J. (2017). Hydrogeologic characteristics and geospatial analysis of water table changes in the alluvium of the lower Arkansas River Valley, southeastern Colorado, 2002, 2008, and 2015. U.S. Geological Survey Scientific Investigations Map 3378, 20 pp. Accessed March 29, 2021 from https://pubs.usgs.gov/sim/3378/sim3378.pdf	Approximated from Fig. 4 of Holmgren (2017); eastern and western reaches separated by John Martin Reservoir.
Lower Arkansas River Eastern Reach	-	Holmberg, M.J. (2017). Hydrogeologic characteristics and geospatial analysis of water table changes in the alluvium of the lower Arkansas River Valley, southeastern Colorado, 2002, 2008, and 2015. U.S. Geological Survey Scientific Investigations Map 3378, 20 pp. Accessed March 29, 2021 from https://pubs.usgs.gov/sim/3378/sim3378.pdf	Approximated from Fig. 4 of Holmgren (2017); eastern and western reaches separated by John Martin Reservoir.
Abobe Creek Aquifer	-	Mustard, M.H., Cain, D. (1984). Hydrology and chemical quality of ground water in Kiowa County, Colorado. U.S. Geological Survey Open File Report 84-1023, 2 plates. Accessed March 29, 2021 from https://pubs.er.usgs.gov/publication/ofr84-1023	Approximated from Plate 1 by Mustard and Cain (1984).
Nussbaum Aquifer	-	Mustard, M.H., Cain, D. (1984). Hydrology and chemical quality of ground water in Kiowa County, Colorado. U.S. Geological Survey Open File Report 84-1023, 2 plates. Accessed March 29, 2021 from https://pubs.er.usgs.gov/publication/ofr84-1023	Approximated from Plate 1 by Mustard and Cain (1984).
Big Sandy-Rush Creek Aquifer	-	Mustard, M.H., Cain, D. (1984). Hydrology and chemical quality of ground water in Kiowa County, Colorado. U.S. Geological Survey Open File Report 84-1023, 2 plates. Accessed March 29, 2021 from	Approximated from Plate 1 by Mustard and Cain (1984).

Aquifer System	Broader System	References			Delineation steps and sources
		https://pubs.er.usgs.gov/publication/ofr81-4023			
Fountain Creek Alluvial Aquifer	-	Radell, M.J., Lewis, M.E., Watts, K.R. (1994). Hydrogeologic characteristics of the alluvial aquifer and adjacent deposits of the Fountain Creek valley, El Paso County, Colorado. U.S. Geological Survey Water Resources Investigations Report 94-4129, 3 maps and 11 cross sections. Accessed March 29, 2021 from https://pubs.er.usgs.gov/publication/wri94-4129	-	-	Approximated from Plate 1 of Radell et al. (1994).
Wind River Basin West	Wind River Basin	Whiteomb, H.A., Lowry, M.E. (1968) Ground-water resources and geology of the Wind River Basin area, central Wyoming. U.S. Geological Survey Hydrologic Atlas 270, 14 pp. Accessed March 29, 2021 from https://pubs.usgs.gov/ha/270/report.pdf	-	-	Approximated from Plate 1 of Whitecomb and Lowry (1968). East-west basin divide approximated south of Boysen Reservoir
Wind River Basin East	Wind River Basin	Whiteomb, H.A., Lowry, M.E. (1968) Ground-water resources and geology of the Wind River Basin area, central Wyoming. U.S. Geological Survey Hydrologic Atlas 270, 14 pp. Accessed March 29, 2021 from https://pubs.usgs.gov/ha/270/report.pdf	-	-	Approximated from Plate 1 of Whitecomb and Lowry (1968). East-west basin divide approximated south of Boysen Reservoir
Northern Green River Basin	-	Bartos, T.T., Hallberg, L.L., Eddy Miller, C.A. (2015). Hydrogeology, groundwater levels, and generalized potentiometric surface map of the Green River Basin lower Tertiary aquifer system, 2010–14, in the northern Green River structural basin, Wyoming. U.S. Geological Survey Scientific Investigations Report 2015–5090, 33 pp. Accessed March 29, 2021 from https://pubs.usgs.gov/sir/2015/5090/sir20155090.pdf	-	-	Approximated from Fig. 4 of Bartos et al. (2015)
Star Valley	-	Walker, E.H. (1965). Ground-water in the upper Star Valley, Wyoming. U.S. Geological Survey Water Supply Paper 1809-C, 33 pp. Accessed March 29, 2021 from https://pubs.usgs.gov/wsp/1809c/report.pdf	-	-	Approximated from Fig. 4 of Walker (1965).
Jackson Hole Aquifer	-	Wright, P.R. (2013). Hydrogeology and water quality in the Snake River alluvial aquifer at Jackson Hole Airport, Jackson,	-	-	Approximated from Fig. 3 of Wright (2013).

Aquifer System	Broader System	References	Delineation steps and sources	
		Wyoming, water years 2011 and 2012; U.S. Geological Survey Scientific Investigations Report 2013-5184, 56 pp. Accessed March 29, 2021 from https://pubs.usgs.gov/sir/2013/5184/pdf/sir2013-5184.pdf		
Hanna and Carbon Basins	-	Daddow, P.D. (1980). Ground-water data for the Hanna and Carbon basins, south-central Wyoming, through 1980. U.S. Geological Survey Open-File Report 85-628, 94 pp. Accessed March 29, 2021 from https://pubs.usgs.gov/of/1985/0628/report.pdf	- - Approximated from Fig. 3 of Daddow (1980).	
Laramie Basin	-	Bradley, E. (1955). Summary of the ground-water resources of the Laramie River drainage basin, Wyoming, and the North Platte River drainage basin from Douglas, Wyoming, to the Wyoming-Nebraska state line. U.S. Geological Survey Open-File Report 55-17, 24 pp. Accessed March 29, 2021 from https://pubs.usgs.gov/of/1955/0017/report.pdf	- - Approximated from Fig. 4 of Bradley (1955)	
Bighorn Basin	-	Tauchen, P., Bartos, T.T., Clarey, K.E., Quillinan, S.A., Hallberg, L.L., Clark, M.L., Thompson, M., Cribb, N., Worman, B., Gracias, T. (2012). Wind/Bighorn River Basin Water Plan Update Groundwater Study Level 1 (2008–2011). Groundwater Determination. Wyoming Water Development Commission Technical Memorandum 397 pp. Accessed March 29, 2021 from https://waterplan.state.wy.us/plan/bighorn/2010/gw-finalrept/gw-finalrept.pdf	Fisher, C.A. (1906). Geology and water resources of the Bighorn Basin, Wyoming. U.S. Geological Survey Professional Paper 53, 97 pp. Accessed November 29, 2021 from https://pubs.usgs.gov/pp/0053/report.pdf	- - Approximate from maps in Chapter 3 of Tauchen et al. (2012) and Plate XII by Fisher (1906)
Lyman Mountain View Area	-	Robinove, C.J., Cummings, T.R. (1963). Ground-water resources and geology of the Lyman Mountain View area, Uinta County, Wyoming. U.S. Geological Survey Water Supply Paper 1669, 48 pp. Accessed March 29, 2021 from https://pubs.usgs.gov/wsp/1669e/report.pdf	- - Approximated from plate 4 of Robinove and Cummings (1963)	
West Bench	-	Carstarphen, C., Patton, T., and LaFave, J.H. (2014). Water levels in the Upper West Bench alluvial aquifer, Red Lodge, Montana. Montana Bureau of Mines and	- - Approximated from Fig. 4 of Carstarphen et al. (2014).	

Aquifer System	Broader System	References	Delineation steps and sources
		Geology Information Pamphlet 8, 8 pp. Accessed March 29, 2021 from http://mbmg.mtech.edu/pdf-publications/ip_8.pdf	
Middle Yellowstone Area	-	Madison, J.P., LaFave, J.I., Patton, T.W., Smith, L.N., Olson, J.N. (2014). Groundwater resources of the Middle Yellowstone River area: Treasure and Yellowstone counties, Montana Part A: Descriptive Overview and Water Quality Data. Montana Bureau of Mines and Geology Montana Ground Water Assessment Atlas 3-A, 82 pp. Accessed March 29, 2021 from http://mbmg.mtech.edu/pdf-publications/gwaa_3.pdf	Approximated from Fig. 10 of Madison et al. (2014).
Gallatin Valley	-	Kendy, E. (2001). Ground-water resources of the Gallatin Local Water Quality District, southwestern Montana. U.S. Geological Survey Fact Sheet 007-01, 4 pp. Accessed March 29, 2021 from https://pubs.usgs.gov/fs/2001/0007/report1.pdf	Approximated from Fig. 1 of Kendy (2001).
Helena Valley Fill	-	Briar, D.W., Madison, J.P. (1992). Hydrogeology of the Helena Valley fill aquifer system, west-central Montana. U.S. Geological Survey Water Resources Investigations Report 92-4023, 97 pp. Accessed March 29, 2021 from https://pubs.usgs.gov/wri/1992/4023/report1.pdf	Approximated from Fig. 1 of Briar and Madison (1992)
Upper Beaverhead Basin	-	Uthman, W., Beck J. (1998). Hydrogeology of the Upper Beaverhead Basin near Dillon, Montana. Montana Bureau of Mines and Geology Open File Report 384, 605 pp. Accessed March 29, 2021 from http://dnr.mt.gov/divisions/water/management/docs/ground-water-studies/hydrogeology_upper_beaverhead_near_dillon.pdf	Approximated from Fig. 1 of Uthman and Beck (1998); northeast boundary extended beyond Fig. 1 outline based on well completion data.
Lower Ruby River Valley	-	Northern Rockies Engineering Inc. (date not provided). Plate B1. Shallow Aquifer Classification of the Lower Ruby Valley, Montana. Accessed March 29, 2021 from https://www.northernrockiesengineering.c	Approximated from plates by Northern Rockies Engineering Inc. (report not dated)

Aquifer System	Broader System	References	Delineation steps and sources	
		content/uploads/2020/03/PlatesB1_B2_Ruby_classification.pdf		
Boulder Valley	-	Butler, J.A. Bobst, A.L. (2017). Hydrogeologic investigation of the Boulder Valley, Jefferson County, Montana. Montana Bureau of Mines and Geology Open File Report 688, 140 pp. Accessed March 29, 2021 from http://mbmg.mtech.edu/pdf-open-files/mbmg688_web.pdf	- - Approximated from Fig. 2 of Butler and Bobst (2017).	
Madison Aquifer	-	LaFave, J. (2011). Quality and Age of Water in the Madison Aquifer, Cascade County, Montana. Montana American Water Resources Association Conference, Session 2. 27 pp. Accessed March 29, 2021 from https://www.montanaawra.org/wp/ppts/2011/session2/5_LaFave_John_j.pdf	Downey, J. S. (1982). Geohydrology of the Madison and associated aquifers in parts of Montana, North Dakota, South Dakota, and Wyoming. U.S. Geological Survey Professional Paper 1273-G, 54 pp. Accessed March 29, 2021 from https://pubs.usgs.gov/pp/1273g/report.pdf	- - While the Madison Aquifer is extensive (spanning multiple states; see Downey (1982)), the relatively small region in central Montana is delineated here as it is "... (most utilized in Cascade Co." (quote LaFave (2011)).
Judith Basin	-	Levings, J. F. (1983). Hydrogeology and simulation of water flow in the Kootenai aquifer of the Judith basin, central Montana. U.S. Geological Survey Water Resources Investigations Report 83-4146, 44 pp. Accessed March 29, 2021 from https://pubs.usgs.gov/wri/1983/4146/report.pdf	- - Approximated from Fig. 4 of Levings (1983).	
Gila Valley	-	Knechtel, M. M., Lohr, E. W. (1938). Geology and ground-water resources of the Valley of Gila River and San Simon Creek, Graham County, Arizona; with a section on the Chemical character of the ground water. U.S. Geological Survey Water Supply Paper 796-F, 67 pp. Accessed March 29, 2021 from https://pubs.usgs.gov/wsp/0796f/report.pdf	- - Approximated from map on page 40 of Knechtel and Lohr (1938).	
San Simon Valley	-	Schwennesen, A.T., Forbes, R.H. (1919). Ground water in San Simon Valley, Arizona and New Mexico. U.S. Geological Survey Water Supply Paper 425, 161 pp. Accessed March 29, 2021 from https://pubs.usgs.gov/wsp/0425a/report.pdf	- - Approximated from plate 1 of Schwennesen and Forbes (1919)	
Animas Valley	-	Schwennesen, A.T., Hare, R.F. (1918). Ground water in the Animas, Playas,	- - Approximated from Plate II of Schwennesen and Hare (1918).	

Aquifer System	Broader System	References	Delineation steps and sources
		Hachita, and San Luis Basins, New Mexico, with analyses of water and soil. U.S. Geological Survey Water Supply Paper 422, 157 pp. Accessed March 29, 2021 from https://pubs.usgs.gov/wsp/0422/report.pdf	
Playas Valley	-	Schwennesen, A.T., Hare, R.F. (1918). Ground water in the Animas, Playas, Hachita, and San Luis Basins, New Mexico, with analyses of water and soil. U.S. Geological Survey Water Supply Paper 422, 157 pp. Accessed March 29, 2021 from https://pubs.usgs.gov/wsp/0422/report.pdf	Approximated from Plate II of Schwennesen and Hare (1918).
Lordsburg Valley	-	Schwennesen, A.T., Hare, R.F. (1918). Ground water in the Animas, Playas, Hachita, and San Luis Basins, New Mexico, with analyses of water and soil. U.S. Geological Survey Water Supply Paper 422, 157 pp. Accessed March 29, 2021 from https://pubs.usgs.gov/wsp/0422/report.pdf	Approximated from Plate II of Schwennesen and Hare (1918).
Central Raton Basin	-	Geldon, A. L. (1989). Ground-water hydrology of the central Raton Basin, Colorado and New Mexico. U.S. Geological Survey Water Supply Paper 2288, 90 pp. Accessed March 29, 2021 from https://pubs.usgs.gov/wsp/2288/report.pdf	Approximated from Fig. 1 of Geldon (1989), with some guidance from Fig. 1 of Watts (2006)
Houston-Galveston Area	Gulf Coast Regional Aquifer System	Weiss, J. S. (1990). Geohydrologic units of the coastal lowlands aquifer system, south-central United States. U.S. Geological Survey Regional aquifer systems analysis, 42 pp. Accessed March 30, 2021 from https://pubs.usgs.gov/pp/1416c/report.pdf	Outer margin of Gulf Coast Regional Aquifer System approximated from Fig. 2 of Weiss (1990).
		Braun, C.L., Ramage, J.K., Shah, S.D. (2019). Status of groundwater-level altitudes and long-term groundwater-level changes in the Chicot, Evangeline, and Jasper aquifers, Houston-Galveston region, Texas, 2019. U.S. Geological Survey Scientific Investigations Report 2019-5089, 18 pp. https://pubs.usgs.gov/sir/2019/5089/sir20195089.pdf	

Aquifer System	Broader System	References		Delineation steps and sources	
Victoria Area	Gulf Coast Regional Aquifer System	Weiss, J. S. (1990). Geohydrologic units of the coastal lowlands aquifer system, south-central United States. U.S. Geological Survey Regional aquifer systems analysis. 42 pp. Accessed March 30, 2021 from https://pubs.usgs.gov/pp/1416c/report.pdf	Marvin, R.F., Shafer, G.H., Dale, O.C., (1962). Groundwater resources of Victoria and Calhoun Counties, Texas. Accessed March 30, 2021 from https://www.twdb.texas.gov/publications/reports/bulletins/doc/Bull.htm/B6202.asp	-	Outer margin of Gulf Coast Regional Aquifer System approximated from Fig. 2 of Weiss (1990). Delineation of central portion of subarea (and regional geology) aided by plate 1a of Marvin et al. (1962). Western margin of subarea approximated along the Nueces River.
Rio Grande Delta	Gulf Coast Regional Aquifer System	Weiss, J. S. (1990). Geohydrologic units of the coastal lowlands aquifer system, south-central United States. U.S. Geological Survey Regional aquifer systems analysis. 42 pp. Accessed March 30, 2021 from https://pubs.usgs.gov/pp/1416c/report.pdf	-	-	Outer margin of Gulf Coast Regional Aquifer System approximated from Fig. 2 of Weiss (1990). Mexico portion of aquifer system delineated based on well completion data and topography.
Lafayette Area	Gulf Coast Regional Aquifer System	Weiss, J. S. (1990). Geohydrologic units of the coastal lowlands aquifer system, south-central United States. U.S. Geological Survey Regional aquifer systems analysis. 42 pp. Accessed March 30, 2021 from https://pubs.usgs.gov/pp/1416c/report.pdf	Louisiana Department of Transportation and Development (2011). Water Resources of Lafayette Parish. U.S. Geological Survey Fact Sheet 2010-3048, 6 pp. Accessed March 30, 2021 from https://pubs.usgs.gov/fs/2010/3048/pdf/FS2010-3048.pdf	-	Outer margin of Gulf Coast Regional Aquifer System approximated from Fig. 2 of Weiss (1990). Detailed study site information for a portion of the subarea available from Louisiana Department of Transportation and Development (2011)
Gonzales-New Orleans Aquifer	Gulf Coast Regional Aquifer System	Weiss, J. S. (1990). Geohydrologic units of the coastal lowlands aquifer system, south-central United States. U.S. Geological Survey Regional aquifer systems analysis. 42 pp. Accessed March 30, 2021 from https://pubs.usgs.gov/pp/1416c/report.pdf	Louisiana Department of Transportation and Development (2014a). Water Resources of St. John the Baptist Parish, Louisiana. U.S. Geological Survey Fact Sheet 2014-3102, 6 pp. Accessed March 30, 2021 from https://pubs.usgs.gov/fs/2014/3102/pdf/fs2014-3102.pdf	Louisiana Department of Transportation and Development (2014b). Water Resources of Orleans Parish, Louisiana. U.S. Geological Survey Fact Sheet 2014-3047, 6 pp. Accessed March 30, 2021 from https://pubs.usgs.gov/fs/2014/3047/pdf/fs2014-3047.pdf	Outer margin of Gulf Coast Regional Aquifer System approximated from Fig. 2 of Weiss (1990). Detailed study site information for a portion of the subarea available from Louisiana Department of Transportation and Development (2014a, b)
Mississippi Delta	Gulf Coast Regional Aquifer System	Weiss, J. S. (1990). Geohydrologic units of the coastal lowlands aquifer system, south-central United States. U.S. Geological Survey Regional aquifer systems analysis. 42 pp. Accessed March 30, 2021 from https://pubs.usgs.gov/pp/1416c/report.pdf	Forak, L.J., Painter, J.A. (2019). Geostatistical estimation of the bottom altitude and thickness of the Mississippi River Valley alluvial aquifer. U.S. Geological Survey Scientific Investigations Map 3426, 2 sheets, Accessed March 31, 2021 from https://pubs.er.usgs.gov/publication/sim3426 .	-	Outer margin of Gulf Coast Regional Aquifer System approximated from Fig. 2 of Weiss (1990). Delta subarea delineated southeast of Lafayette Area, south of deep wells (from completion reports) near Baton Rouge, and south of Gonzales-New Orleans aquifer area. Torak and Painter (2019) provide helpful alluvial thickness data for the delta region and also the Mississippi River Valley Alluvial Aquifer extending

Aquifer System	Broader System	References	Delineation steps and sources
Southern Hills	Gulf Coast Regional Aquifer System	Weiss, J. S. (1990). Geohydrologic units of the coastal lowlands aquifer system, south-central United States. U.S. Geological Survey Regional aquifer systems analysis. 42 pp. Accessed March 30, 2021 from https://pubs.usgs.gov/pp/1416c/report.pdf	north of this delineated subregion (connecting to the Central Mississippi Embayment); alluvium in this delineated area can be up to 120 m thick (Torak and Painter, 2019). Margin of Gulf Coast Regional Aquifer System approximated from Fig. 2 of Weiss (1990); northwestern boundary is Central Mississippi Embayment. Eastern border approximated from Figs. 1 and 2 of Buono (1983) and the Pearl River. Shallow aquifer ("Citronelle aquifers") information from Boswell (1979).
Catahoula Area	Gulf Coast Regional Aquifer System	Weiss, J. S. (1990). Geohydrologic units of the coastal lowlands aquifer system, south-central United States. U.S. Geological Survey Regional aquifer systems analysis. 42 pp. Accessed March 30, 2021 from https://pubs.usgs.gov/pp/1416c/report.pdf	Northern margin of Gulf Coast Regional Aquifer System approximated from Fig. 2 of Weiss (1990). Hydrostratigraphy for central portion of subarea detailed by Halford and Barber (1995). Eastern margin approximated along portions of the Chickasawhay River.
Alabama Coastal Lowlands	Gulf Coast Regional Aquifer System	Mallory, M. J. (1993). Hydrogeology of the Southeastern Coastal Plain aquifer system in parts of eastern Mississippi and western Alabama. U.S. Geological Survey Professional Paper 1410-G, 66 pp. Accessed March 31, 2021 from https://pubs.usgs.gov/pp/1410g/report.pdf	Eastern margin approximated along the Conecuh and Escambia Rivers. This area was not included in Floridan Aquifer System, as some local hydrogeological reports in its eastern margins (e.g. Cantonment in northwest Florida) suggest the Floridan aquifer itself may be quite deep as "(t)he Pensacola Clay has been reached by only a few wells in the area" (quote from Trapp (1973)); and the Pensacola Clay overlies what has been deemed the "Floridan Aquifer System" (see Fig. 1 of Pratt et al. (1996)). The subarea contains an area that has been delineated as being part of both the Floridan and

Aquifer System	Broader System	References	Delineation steps and sources		
			the Gulf Coastal Plain (see overlapping areas mapped in Fig. 4 of Mallory (1993)). Fig. 8 of Barraclough and Marsh (1962) also shows that the Upper Floridan Aquifer is more than 300 m deep in Baldwin County (Alabama) as the primary units of the Floridan do not immediately appear to be widely accessed, the subarea was included in the Gulf Coast Regional Aquifer rather than the Floridan Aquifer System.		
Dougherty Plain and Marianna Lowlands	Floridan Aquifer System	Williams, L. J., Kuniansky, E. L. (2016). Revised hydrogeologic framework of the Floridan aquifer system in Florida and parts of Georgia, Alabama, and South Carolina. U.S. Geological Survey Professional Paper 1807, 140 pp. Accessed March 31, 2021 from https://pubs.usgs.gov/pp/1807/pdf/pp1807.pdf	Clark, W. Z., Zisa, A. C. (1976). Physiographic map of Georgia: Georgia Department of Natural Resources, 1 sheet. Accessed March 31, 2021 from https://epd.georgia.gov/document/publication/em-4-physiographic-map-georgia-12000000-1988/download	-	Approximated from Fig. 21 of Williams and Kuniansky (2016). Georgia portion of area also provided by Clark and Zisa (1976).
Southern Pine Hills	Floridan Aquifer System	Barraclough, J. T., Marsh, O. T. (1962). Aquifers and quality of ground water along the Gulf Coast of western Florida. U.S. Geological Survey Report of Investigations No. 62, 41 pp. Accessed March 31, 2021 from https://ufdcimages.uflib.ufl.edu/UF/00/00/12/16/00001/UF00001216.pdf	-	-	Approximated from Fig. 1 of Barraclough and Marsh (1962)
Tifton Upland	Floridan Aquifer System	Davis, H. (1996). Hydrogeologic Investigation and Simulation of Ground-Water Flow in the Upper Floridan Aquifer of North-Central Florida and Southwestern Georgia and Delineation of Contributing Areas for Selected City of Tallahassee, Florida, Water-Supply Wells. U.S. Geological Survey Water-Resources Investigations Report 95-4296, 61 pp. Accessed March 31, 2021 from https://fl.water.usgs.gov/PDF_files/wri95_4296_davis.pdf	Clark, W. Z., Zisa, A. C. (1976). Physiographic map of Georgia: Georgia Department of Natural Resources, 1 sheet. Accessed March 31, 2021 from https://epd.georgia.gov/document/publication/em-4-physiographic-map-georgia-12000000-1988/download	Brooks, H. K. (1981). Physiographic divisions of Florida. Report for the Florida Cooperative Extension Service Institute of Food and Agricultural Sciences, University of Florida, 12 pp.	Approximated from Fig. 2 of Davis (1996) and the physiographic provinces for Georgia and Florida by Clark and Zisa (1976) and Brooks (1981)
Apalachicola Delta	Floridan Aquifer System	Davis, H. (1996). Hydrogeologic Investigation and Simulation of Ground-Water Flow in the Upper Floridan Aquifer of North-Central Florida and Southwestern Georgia and Delineation of	Clark, W. Z., Zisa, A. C. (1976). Physiographic map of Georgia: Georgia Department of Natural Resources, 1 sheet. Accessed March 31, 2021 from	Brooks, H. K. (1981). Physiographic divisions of Florida. Report for the Florida Cooperative Extension Service Institute of Food and	Approximated from Fig. 2 of Davis (1996) and the physiographic provinces for Georgia and Florida by Clark and Zisa (1976) and Brooks (1981)

Aquifer System	Broader System	References			Delineation steps and sources
		Contributing Areas for Selected City of Tallahassee, Florida, Water Supply Wells. U.S. Geological Survey Water Resources investigations Report 95-4296, 61 pp. Accessed March 31, 2021 from https://fl.water.usgs.gov/PDF_files/wri95_4296_davis.pdf	https://epd.georgia.gov/document/publication/em-4-physiographic-map-georgia-12000000-1988/download	Agricultural Sciences, University of Florida, 12 pp.	
Ocala Uplift	Floridan Aquifer System	Davis, H. (1996). Hydrogeologic Investigation and Simulation of Ground-Water Flow in the Upper Floridan Aquifer of North-Central Florida and Southwestern Georgia and Delineation of Contributing Areas for Selected City of Tallahassee, Florida, Water Supply Wells. U.S. Geological Survey Water Resources investigations Report 95-4296, 61 pp. Accessed March 31, 2021 from https://fl.water.usgs.gov/PDF_files/wri95_4296_davis.pdf	Clark, W. Z., Zisa, A. C. (1976) Physiographic map of Georgia: Georgia Department of Natural Resources, 1 sheet. Accessed March 31, 2021 from https://epd.georgia.gov/document/publication/em-4-physiographic-map-georgia-12000000-1988/download	Brooks, H. K. (1981). Physiographic divisions of Florida. Report for the Florida Cooperative Extension Service Institute of Food and Agricultural Sciences, University of Florida, 12 pp.	Approximated from Fig. 2 of Davis (1996) and the physiographic provinces for Georgia and Florida by Clark and Zisa (1976) and Brooks (1981)
Southwestern Flatwoods	Floridan Aquifer System	Williams, L.J., Kuniandy, E.L. (2016). Revised hydrogeologic framework of the Floridan aquifer system in Florida and parts of Georgia, Alabama, and South Carolina. U.S. Geological Survey Professional Paper 1807, 140 pp. Accessed March 31, 2021 from https://pubs.usgs.gov/pp/1807/pdf/pp1807.pdf	Brooks, H. K. (1981). Physiographic divisions of Florida. Report for the Florida Cooperative Extension Service Institute of Food and Agricultural Sciences, University of Florida, 12 pp.	-	Approximated from physiographic provinces of Florida outlined by Brooks (1981). Broader Florida Aquifer System from Williams and Kuniandy (2016).
Central Lake Area	Floridan Aquifer System	Williams, L.J., Kuniandy, E.L. (2016). Revised hydrogeologic framework of the Floridan aquifer system in Florida and parts of Georgia, Alabama, and South Carolina. U.S. Geological Survey Professional Paper 1807, 140 pp. Accessed March 31, 2021 from https://pubs.usgs.gov/pp/1807/pdf/pp1807.pdf	Brooks, H. K. (1981). Physiographic divisions of Florida. Report for the Florida Cooperative Extension Service Institute of Food and Agricultural Sciences, University of Florida, 12 pp.	-	Approximated from physiographic provinces of Florida outlined by Brooks (1981). Broader Florida Aquifer System from Williams and Kuniandy (2016).
Eastern Flatwoods Southshores	Floridan Aquifer System	Williams, L.J., Kuniandy, E.L. (2016). Revised hydrogeologic framework of the Floridan aquifer system in Florida and parts of Georgia, Alabama, and South Carolina. U.S. Geological Survey Professional Paper 1807, 140 pp. Accessed March 31, 2021 from https://pubs.usgs.gov/pp/1807/pdf/pp1807.pdf	Brooks, H. K. (1981). Physiographic divisions of Florida. Report for the Florida Cooperative Extension Service Institute of Food and Agricultural Sciences, University of Florida, 12 pp.	-	Approximated from physiographic provinces of Florida outlined by Brooks (1981). Broader Florida Aquifer System from Williams and Kuniandy (2016). Eastern Flatwoods as defined by Brooks (1981) was split at Lake Harney into two areas north and south of

Aquifer System	Broader System	References		Delineation steps and sources
Eastern Flatwoods Northshores	Floridan Aquifer System	Williams, L.J., Kuniansky, E.L. (2016). Revised hydrogeologic framework of the Floridan aquifer system in Florida and parts of Georgia, Alabama, and South Carolina. U.S. Geological Survey Professional Paper 1807, 140 pp. Accessed March 31, 2021 from https://pubs.usgs.gov/pp/1807/pdf/pp1807.pdf	Brooks, H. K. (1981). Physiographic divisions of Florida. Report for the Florida Cooperative Extension Service Institute of Food and Agricultural Sciences, University of Florida, 12 pp.	- Lake Harney ('northshores' and 'southshores'); Approximated from physiographic provinces of Florida outlined by Brooks (1981). Broader Florida Aquifer System from Williams and Kuniansky (2016). Eastern Flatwoods as defined by Brooks (1981) was split at Lake Harney into two areas north and south of Lake Harney ('northshores' and 'southshores').
Vidalia Upland	Floridan Aquifer System	Williams, L.J., Kuniansky, E.L. (2016). Revised hydrogeologic framework of the Floridan aquifer system in Florida and parts of Georgia, Alabama, and South Carolina. U.S. Geological Survey Professional Paper 1807, 140 pp. Accessed March 31, 2021 from https://pubs.usgs.gov/pp/1807/pdf/pp1807.pdf	Clark, W. Z., Zisa, A. C. (1976). Physiographic map of Georgia: Georgia Department of Natural Resources, 1 sheet. Accessed March 31, 2021 from https://epd.georgia.gov/document/publication/sm-4-physiographic-map-georgia-12000000-1988/download	- Approximated from physiographic provinces of Georgia outlined by Clark and Zisa (1976). Broader Florida Aquifer System from Williams and Kuniansky (2016). Northeastern boundary set along the Savannah River.
Bacon Terrace	Floridan Aquifer System	Williams, L.J., Kuniansky, E.L. (2016). Revised hydrogeologic framework of the Floridan aquifer system in Florida and parts of Georgia, Alabama, and South Carolina. U.S. Geological Survey Professional Paper 1807, 140 pp. Accessed March 31, 2021 from https://pubs.usgs.gov/pp/1807/pdf/pp1807.pdf	Clark, W. Z., Zisa, A. C. (1976). Physiographic map of Georgia: Georgia Department of Natural Resources, 1 sheet. Accessed March 31, 2021 from https://epd.georgia.gov/document/publication/sm-4-physiographic-map-georgia-12000000-1988/download	- Approximated from physiographic provinces of Georgia outlined by Clark and Zisa (1976). Broader Florida Aquifer System from Williams and Kuniansky (2016).
Okefenokee Basin	Floridan Aquifer System	Williams, L.J., Kuniansky, E.L. (2016). Revised hydrogeologic framework of the Floridan aquifer system in Florida and parts of Georgia, Alabama, and South Carolina. U.S. Geological Survey Professional Paper 1807, 140 pp. Accessed March 31, 2021 from https://pubs.usgs.gov/pp/1807/pdf/pp1807.pdf	Clark, W. Z., Zisa, A. C. (1976). Physiographic map of Georgia: Georgia Department of Natural Resources, 1 sheet. Accessed March 31, 2021 from https://epd.georgia.gov/document/publication/sm-4-physiographic-map-georgia-12000000-1988/download	- Approximated from physiographic provinces of Georgia outlined by Clark and Zisa (1976). Broader Florida Aquifer System from Williams and Kuniansky (2016).
Upper Coastal Plain	Floridan Aquifer System	Williams, L.J., Kuniansky, E.L. (2016). Revised hydrogeologic framework of the Floridan aquifer system in Florida and parts of Georgia, Alabama, and South Carolina. U.S. Geological Survey Professional Paper 1807, 140 pp. Accessed March 31, 2021 from https://pubs.usgs.gov/pp/1807/pdf/pp1807.pdf	Aucott, W.R. (1996). Hydrology of the Southeastern Coastal Plain Aquifer System in South Carolina and Parts of Georgia and North Carolina. U.S. Geological Survey Professional Paper 1410-E, 95 pp. Accessed March 31, 2021 from https://pubs.usgs.gov/pp/1410e/report.pdf	- Broader Florida Aquifer System from Williams and Kuniansky (2016). Approximate divide between the Upper and Lower Coastal Plain from Fig. 1 of Aucott (1996).

Aquifer System	Broader System	References			Delineation steps and sources
Lower Coastal Plain	Floridan Aquifer System	Williams, L.J., Kuniansky, E.L. (2016). Revised hydrogeologic framework of the Floridan aquifer system in Florida and parts of Georgia, Alabama, and South Carolina. U.S. Geological Survey Professional Paper 1807, 140 pp. Accessed March 31, 2021 from https://pubs.usgs.gov/pp/1807/pdf/pp1807.pdf	Aucott, W.R. (1996). Hydrology of the Southeastern Coastal Plain Aquifer System in South Carolina and Parts of Georgia and North Carolina. U.S. Geological Survey Professional Paper 1410-E, 95 pp. Accessed March 31, 2021 from https://pubs.usgs.gov/pp/1410e/report.pdf	-	Broader Florida Aquifer System from Williams and Kuniansky (2016). Approximate divide between the Upper and Lower Coastal Plain from Fig. 1 of Aucott (1996).
Sea Island	Floridan Aquifer System	Williams, L.J., Kuniansky, E.L. (2016). Revised hydrogeologic framework of the Floridan aquifer system in Florida and parts of Georgia, Alabama, and South Carolina. U.S. Geological Survey Professional Paper 1807, 140 pp. Accessed March 31, 2021 from https://pubs.usgs.gov/pp/1807/pdf/pp1807.pdf	Brooks, H. K. (1981). Physiographic divisions of Florida. Report for the Florida Cooperative Extension Service Institute of Food and Agricultural Sciences, University of Florida, 12 pp.	Clark, W. Z., Zisa, A. C. (1976). Physiographic map of Georgia. Georgia Department of Natural Resources, 1 sheet. Accessed March 31, 2021 from https://epd.georgia.gov/document/publication/sm-4-physiographic-map-georgia-42000000-1988/download	Approximated from physiographic provinces for Georgia and Florida by Clark and Zisa (1976) and Brooks (1981). Broader Florida Aquifer System from Williams and Kuniansky (2016).
Intermediate Aquifer	Floridan Aquifer System	Miller, J.A. (1990). Ground Water Atlas of the United States: Segment 6, Alabama, Florida, Georgia, South Carolina. U.S. Geological Survey Hydrologic Atlas 730-G, 30 pp. Accessed April 5, 2021 from https://www.nrc.gov/docs/ML1706/ML17060B027.pdf	Knochenmus, L.A. (2006). Regional Evaluation of the Hydrogeologic Framework, Hydraulic Properties, and Chemical Characteristics of the Intermediate Aquifer System Underlying Southern West-Central Florida. U.S. Geological Survey Scientific Investigations Report 2006-5013, 62 pp. Accessed April 5, 2021 from https://pubs.usgs.gov/sir/2006/5013/pdf/2006-5013.pdf	-	Approximated Fig. 4 of Knochenmus (2006); aquifer outline also displayed in Figs. 9 and 42 of Miller (1990)
Castle Hayne Aquifer	-	Lyke, W.L., Coble, R.W. (1987). Regional study of the Castle Hayne Aquifer of eastern North Carolina. U.S. Geological Survey Open-File Report 87-571, 2 pp. Accessed April 1, 2021 from https://pubs.usgs.gov/of/1987/0571/report.pdf	Elmers, J.L., Daniel III, C.C., Coble, R.W. (1994). Hydrogeology and simulation of ground-water flow at U.S. Marine Corps Air Station, Cherry Point, North Carolina, 1987-90. Water Resources Investigations Report 94-4186, 81 pp. Accessed April 1, 2021 from https://pubs.usgs.gov/wri/1994/4186/report.pdf	Winner Jr., M.D., Coble, R.W. (1989). Hydrogeologic framework of the North Carolina Coastal Plain aquifer system. U.S. Geological Survey Report 87-690, 167 pp. Accessed April 1, 2021 from https://pubs.usgs.gov/of/1987/0690/report.pdf	Approximated from Fig. 4 of Lyke and Coble (1987) and Fig. 14 of Winner Jr. and Coble (1989). Eastern limit of Castle Hayne is extended farther than shown in Fig. 14 of Winner Jr. and Coble (1989) as local reports near Cherry Point detail the importance of the Castle Hayne Aquifer—see Elmers et al. (1994).
Peedee and Black Creek and Cape Fear Aquifers	-	Winner Jr., M.D., Coble, R.W. (1989). Hydrogeologic framework of the North Carolina Coastal Plain aquifer system. U.S. Geological Survey Report 87-690, 167 pp. Accessed April 1, 2021 from https://pubs.usgs.gov/of/1987/0690/report.pdf	Harden, S. L., Fine, J. M., Spruill, T. B. (2003). Hydrogeology and ground-water quality of Brunswick County, North Carolina. U.S. Geological Survey, Water Resources Investigations Report 03-4051, 98 pp. Accessed April 1, 2021 from	-	Approximated from Fig. 2 of Winner Jr. and Coble (1989) and Plate 9 of Harden et al. (2003).

Aquifer System	Broader System	References			Delineation steps and sources
				https://pubs.usgs.gov/wri/2003/4054/wri20034054.pdf	
Stockton Plateau	Edwards-Trinity Aquifer System	Bruun, B., Jackson, K., Lake, P., Walker, J. (2016). Texas Aquifers Study. Texas Water Development Board Report. 336 pp. Accessed April 1, 2021 from https://www.twdb.texas.gov/groundwater/docs/studies/TexasAquifersStudy_2016.pdf#page=89	Barker, R. A., Ardis, A. F. (1996). Hydrogeological framework of the Edwards-Trinity aquifer system, west-central Texas. U.S. Geological Survey Professional Paper 1421-B, 76 pp. Accessed April 1, 2021 from https://pubs.usgs.gov/pp/1421b/report.pdf	Ryder, P. (1996). Ground Water Atlas of the United States Segment 4 Oklahoma, Texas. Hydrologic Investigations Atlas 730-E, 32 pp. Accessed November 29, 2021 from https://pubs.usgs.gov/ha/730e/report.pdf	Approximated from Fig. 6-11 by Bruun et al. (2016), Fig. 84 by Ryder (1996), and Fig. 3 by Barker and Ardis (1996).
Edwards Plateau	Edwards-Trinity Aquifer System	Bruun, B., Jackson, K., Lake, P., Walker, J. (2016). Texas Aquifers Study. Texas Water Development Board Report. 336 pp. Accessed April 1, 2021 from https://www.twdb.texas.gov/groundwater/docs/studies/TexasAquifersStudy_2016.pdf#page=89	Barker, R. A., Ardis, A. F. (1996). Hydrogeological framework of the Edwards-Trinity aquifer system, west-central Texas. U.S. Geological Survey Professional Paper 1421-B, 76 pp. Accessed April 1, 2021 from https://pubs.usgs.gov/pp/1421b/report.pdf	-	Approximated from Fig. 6-11 by Bruun et al. (2016) and Fig. 3 by Barker and Ardis (1996).
Balcones Fault Zone	Edwards-Trinity Aquifer System	Bruun, B., Jackson, K., Lake, P., Walker, J. (2016). Texas Aquifers Study. Texas Water Development Board Report. 336 pp. Accessed April 1, 2021 from https://www.twdb.texas.gov/groundwater/docs/studies/TexasAquifersStudy_2016.pdf#page=89	Barker, R. A., Ardis, A. F. (1996). Hydrogeological framework of the Edwards-Trinity aquifer system, west-central Texas. U.S. Geological Survey Professional Paper 1421-B, 76 pp. Accessed April 1, 2021 from https://pubs.usgs.gov/pp/1421b/report.pdf	-	Approximated from Fig. 6-6 by Bruun et al. (2016) and Fig. 3 by Barker and Ardis (1996).
Hill Country	Edwards-Trinity Aquifer System	Bruun, B., Jackson, K., Lake, P., Walker, J. (2016). Texas Aquifers Study. Texas Water Development Board Report. 336 pp. Accessed April 1, 2021 from https://www.twdb.texas.gov/groundwater/docs/studies/TexasAquifersStudy_2016.pdf#page=89	Barker, R. A., Ardis, A. F. (1996). Hydrogeological framework of the Edwards-Trinity aquifer system, west-central Texas. U.S. Geological Survey Professional Paper 1421-B, 76 pp. Accessed April 1, 2021 from https://pubs.usgs.gov/pp/1421b/report.pdf	-	Approximated from Fig. 6-11 by Bruun et al. (2016) and Fig. 3 by Barker and Ardis (1996).
Trinity Aquifer System	Edwards-Trinity Aquifer System	Bruun, B., Jackson, K., Lake, P., Walker, J. (2016). Texas Aquifers Study. Texas Water Development Board Report. 336 pp. Accessed April 1, 2021 from https://www.twdb.texas.gov/groundwater/docs/studies/TexasAquifersStudy_2016.pdf#page=89	Gordon, C.H. (1913). Geology and underground waters of the Wichita region, north-central Texas. U.S. Geological Survey Water Supply Paper 347, 90 pp. Accessed April 7, 2021 from https://pubs.usgs.gov/wsp/0347/report.pdf	Ryder, P. (1996). Ground Water Atlas of the United States Segment 4 Oklahoma, Texas. Hydrologic Investigations Atlas 730-E, 32 pp. Accessed November 29, 2021 from https://pubs.usgs.gov/ha/730e/report.pdf	Approximated from Fig. 6-37 by Bruun et al. (2016) and Fig. 114 by Ryder (1996). Delineated area includes the northeast portion of the Balcones Fault Zone following the major aquifer delineation of in map on page vii of Bruun et al. (2016). Local hydrogeology in northern area of aquifer are detailed in Plate 4 of Gordon (1913).
Pecos Valley	-	Bruun, B., Jackson, K., Lake, P., Walker, J. (2016). Texas Aquifers Study. Texas	-	-	Approximated from Fig. 6-28 by Bruun et al. (2016). New Mexico

Aquifer System	Broader System	References	Delineation steps and sources
		Water Development Board Report, 336 pp. Accessed April 1, 2021 from https://www.twdb.texas.gov/groundwater/docs/studies/TexasAquifersStudy_2016.pdf#page=89	portion of aquifer approximated on the basis of well completion records.
Seymour-Blaine Aquifer System	-	Bruun, B., Jackson, K., Lake, P., Walker, J. (2016). Texas Aquifers Study. Texas Water Development Board Report, 336 pp. Accessed April 1, 2021 from https://www.twdb.texas.gov/groundwater/docs/studies/TexasAquifersStudy_2016.pdf#page=89	Approximated from Fig. 6-33 by Bruun et al. (2016) and map by Chastain-Howley et al. (2004) available at https://www.twdb.texas.gov/groundwater/models/gam/symr/symr.asp (Accessed April 5, 2021).
Garber-Wellington Aquifer	-	Mashburn, S.L., Ryfer, D.W., Neel, C.R., Smith, S.J., and Correll, J.S. (2014) Hydrogeology and simulation of groundwater flow in the Central Oklahoma (Garber-Wellington) Aquifer, Oklahoma, 1987 to 2009, and simulation of available water in storage, 2010–2059. U.S. Geological Survey Scientific Investigations Report 2013–5219, 92 pp. Accessed April 5, 2021 from https://pubs.usgs.gov/sir/2013/5219/pdf/sir20135219_v2.0.pdf	Approximated from Fig. 4 of Mashburn et al. (2014)
Arbuckle-Simpson Aquifer	-	Christenson, S., Osborn, N.I., Neel, C.R., Faith, J.R., Blome, C.D., Puckette, James, Pantea, M.P. (2011). Hydrogeology and simulation of groundwater flow in the Arbuckle-Simpson aquifer, south-central Oklahoma. U.S. Geological Survey Scientific Investigations Report 2011–5029, 104 pp. Accessed April 5, 2021 from https://pubs.usgs.gov/sir/2011/5029/SIR2011-5029.pdf	Approximated from Fig. 1 of Christenson et al. (2011)
Tillman Terrace	-	Osborn, N.I. (2002). Update of the Hydrologic Survey of the Tillman Terrace Groundwater Basin, Southwestern Oklahoma. Oklahoma Water Resources Board Technical Report GW2002-1, 24 pp., Accessed April 5, 2021 from https://www.owrb.ok.gov/studies/reports/reports_pdf/tillman_update.pdf	Approximated from well completion data and Fig. 1 of Osborn (2002).
Vamoosa-Ada Aquifer	-	D'Lugoez, J.J., McClaffin, R.C. (1986). Geohydrology of the Vamoosa-Ada aquifer east-central Oklahoma with a	Approximated from plate 1 of D'Lugoez and McClaffin (1986)

Aquifer System	Broader System	References			Delineation steps and sources
		section on chemical quality of water. U.S. Geological Survey Circular 87, 48 pp. Accessed April 5, 2021 from http://www.ogs-ou.edu/pubsscanned/Circulars/circular87mm.pdf			
Cimarron Basin	-	Adams, G.P., Bergman, D.L. (1996). Geohydrology of alluvium and terrace deposits of the Cimarron River from freedom to Guthrie, Oklahoma. U.S. Geological Survey Water Resources Investigations Report 95-4066, 63 pp. Accessed April 5, 2021 from https://pubs.usgs.gov/wri/1995/4066/report.pdf	-	-	Approximated from Fig. 1 of Adams and Bergman (1996).
Enid Isolated Terrace Aquifer	-	Becker, C.J., Runkle, D., Rea, A. (1997a). Digital Data Sets that Describe Aquifer Characteristics of the Enid Isolated Terrace Aquifer in Northwestern Oklahoma. U.S. Geological Survey Open File Report, 96-450. Accessed April 5, 2021 from https://pubs.usgs.gov/of/1996/of96-450/	-	-	Approximated from map by Becker et al. (1997a) available at https://pubs.usgs.gov/of/1996/of96-450/
Antlers Aquifer	-	Hart Jr., Davis, R.E. (1981). Geohydrology of the Antlers aquifer (Cretaceous), southeastern Oklahoma. U.S. Geological Survey Circular 81, 38 pp. Accessed April 5, 2021 from http://www.ogs-ou.edu/pubsscanned/Circulars/circular81mm.pdf	-	-	Approximated from Fig. 1 of Hart and Davis (1981)
Rush Springs Aquifer	-	Becker, M.F., Runkle, D.L. (1998). Hydrogeology, water quality, and geochemistry of the Rush Springs aquifer, western Oklahoma. U.S. Geological Survey Water Resources Investigations Report 98-4081, 44 pp. Accessed April 5, 2021 from https://pubs.usgs.gov/wri/1998/4081/report.pdf	-	-	Approximated from Fig. 1 of Becker and Runkle (1998)
Elk City Aquifer	-	Becker, C.J., Runkle, D., Rea, A. (1997b). Digital data sets that describe aquifer characteristics of the Elk City Aquifer in western Oklahoma. U.S. Geological Survey Open File Report 96-449. Accessed April 5, 2021 from https://pubs.usgs.gov/of/1996/of96-449/	-	-	Approximated from map by Becker et al. (1997b) displayed at: https://pubs.usgs.gov/of/1996/of96-449/ (accessed April 5, 2021)
Canadian River Alluvial Aquifer	-	Ellis, J.H., Mashburn, S.L., Graves, G.M., Peterson, S.M., Smith, S.J., Fuhrig, L.T., Wagner, D.L., Sanford, J.E. (2017).	-	-	Approximated from Fig. 1 of Ellis et al. (2017).

Aquifer System	Broader System	References	Delineation steps and sources	
		Hydrogeology and simulation of groundwater flow and analysis of projected water use for the Canadian River alluvial aquifer, western and central Oklahoma. U.S. Geological Survey Scientific Investigations Report 2016-5180, 64 pp. Accessed April 5, 2021 from https://pubs.usgs.gov/sir/2016/5180/sir20165180.pdf		
Red River Aquifer	-	Smith, S.J., Ellis, J.H., Paizis, N.C., Becker, C.J., Wagner, D.L., Correll, J.S., Hernandez, R.J. (2021). Hydrogeology and model-simulated groundwater availability in the Salt Fork Red River aquifer, southwestern Oklahoma, 1980–2015. U.S. Geological Survey Scientific Investigations Report 2021-5003, 85 pp. Accessed April 5, 2021 from https://pubs.usgs.gov/sir/2021/5003/sir20215003.pdf	Smith, S.J., Ellis, J.H., Wagner, D.L., Peterson, S.M. (2017). Hydrogeology and simulated groundwater flow and availability in the North Fork Red River aquifer, southwest Oklahoma, 1980–2013. U.S. Geological Survey Scientific Investigations Report 2017-5098, 107 pp. Accessed April 5, 2021 from https://pubs.usgs.gov/sir/2017/5098/sir20175098.pdf	Approximated from Fig. 4 of Smith et al. (2017) and Fig. 1 of Smith et al. (2021).
Pearl and Chattahoochee Aquifer System	-	Miller, J.A. (1990). Ground Water Atlas of the United States: Segment 6, Alabama, Florida, Georgia, South Carolina. U.S. Geological Survey Hydrologic Atlas 730-G, 30 pp. Accessed April 5, 2021 from https://www.nrc.gov/docs/ML1706/ML17060B027.pdf	-	Approximated from Fig. 73 of Miller (1990)
Biscayne Aquifer	-	Wacker, M.A., Cunningham, K.J., Williams, J.H. (2014). Geologic and hydrogeologic frameworks of the Biscayne aquifer in central Miami-Dade County, Florida. U.S. Geological Survey Scientific Investigations Report 2014-5138, 66 pp. Accessed March 31, 2021 from https://pubs.usgs.gov/sir/2014/5138/pdf/sir2014-5138.pdf	-	Approximated from Fig. 4 of Wacker et al. (2014).
Connecticut Valley	-	Olcott, P.G. (1995). Ground Water Atlas of the United States: Segment 12, Connecticut, Maine, Massachusetts, New Hampshire, New York, Rhode Island, Vermont. U.S. Geological Survey Hydrologic Atlas 730-M, 30 pp. Accessed April 14, 2021 from https://pubs.usgs.gov/hal/730m/report.pdf	Connecticut Department of Energy and Environmental Protection (2021). Webpage entitled "Overview of the Ground Water Flow System in Connecticut". Accessed April 14, 2021 from https://portal.ct.gov/DEEP/Aquifer-Protection-and-Groundwater/Ground-Water/Ground-Water-Flow-System-in-Connecticut	Approximated from map displayed at top of webpage by Connecticut Department of Energy and Environmental Protection (2021) and also from Fig. 102 of Olcott (1995).

Aquifer System	Broader System	References			Delineation steps and sources
Southern Piedmont Upland	Piedmont Plateau	Geological Survey of Alabama (2018). Assessment of groundwater resources in Alabama, 2010–16. Geological Survey of Alabama Bulletin 186, 462 pp. Accessed April 8, 2021 from https://www.gsa.state.al.us/img/Groundwater/docs/assessment/00_B186_StatewideAssessment_Print_Document.pdf	Miller, J.A. (1990). Ground Water Atlas of the United States: Segment 6, Alabama, Florida, Georgia, South Carolina. U.S. Geological Survey Hydrologic Atlas 730-G, 30 pp. Accessed April 5, 2021 from https://www.nrc.gov/docs/ML1706/ML17060B027.pdf	-	Approximated from Fig. 4 of Geological Survey of Alabama (2018) and Fig. 3 of Miller (1990). Northeastern margin of area approximated along the Savannah River.
Southcentral Piedmont Upland	Piedmont Plateau	Miller, J.A. (1990). Ground Water Atlas of the United States: Segment 6, Alabama, Florida, Georgia, South Carolina. U.S. Geological Survey Hydrologic Atlas 730-G, 30 pp. Accessed April 5, 2021 from https://www.nrc.gov/docs/ML1706/ML17060B027.pdf	-	-	Approximated from Fig. 3 of Miller (1990). Northeastern margin of area approximated along the South Yadkin and Pee Dee Rivers.
Central Piedmont Upland	Piedmont Plateau	Miller, J.A. (1990). Ground Water Atlas of the United States: Segment 6, Alabama, Florida, Georgia, South Carolina. U.S. Geological Survey Hydrologic Atlas 730-G, 30 pp. Accessed April 5, 2021 from https://www.nrc.gov/docs/ML1706/ML17060B027.pdf	-	-	Approximated from Fig. 3 of Miller (1990). Northeastern margin of area approximated along the Hardware and James Rivers.
Northeastern Piedmont Upland	Piedmont Plateau	Miller, J.A. (1990). Ground Water Atlas of the United States: Segment 6, Alabama, Florida, Georgia, South Carolina. U.S. Geological Survey Hydrologic Atlas 730-G, 30 pp. Accessed April 5, 2021 from https://www.nrc.gov/docs/ML1706/ML17060B027.pdf	-	-	Approximated from Fig. 3 of Miller (1990). Northeastern margin of area approximated along the Susquehanna River.
Northern Piedmont Upland	Piedmont Plateau	Miller, J.A. (1990). Ground Water Atlas of the United States: Segment 6, Alabama, Florida, Georgia, South Carolina. U.S. Geological Survey Hydrologic Atlas 730-G, 30 pp. Accessed April 5, 2021 from https://www.nrc.gov/docs/ML1706/ML17060B027.pdf	-	-	Approximated from Fig. 3 of Miller (1990).
New Jersey Coastal Plain	North Atlantic Coastal Plain	Gill, H.E., Farlekas, G.M. (1976). Geohydrologic maps of the Potomac-Raritan-Magothy aquifer system in the New Jersey Coastal Plain. U.S. Geological Survey Hydrologic Atlas 557, 2 plates. Accessed April 1, 2021 from https://pubs.er.usgs.gov/publication/ha557	Gordon, A.D., Carleton, G.B., Rosman, B. (2021). Water-level conditions in the confined aquifers of the New Jersey Coastal Plain, 2013. U.S. Geological Survey Scientific Investigations Report 2019–5146, 104 p., 9 pl., Accessed April 1, 2021 from https://pubs.usgs.gov/sir/2019/5146/sir20195146.pdf	-	Approximated from Plate 1 of Gill et al. (1976) and Fig. 1 of Gordon et al. (2021).

Aquifer System	Broader System	References			Delineation steps and sources
Delmarva Peninsula	North Atlantic Coastal Plain	Bachman, L.J., Shedlock, R.J., Phillips, P.J. (1987). Ground-water quality assessment of the Delmarva Peninsula, Delaware, Maryland, and Virginia. U.S. Geological Survey Open-File Report 87-112. Accessed April 1, 2021 from https://pubs.usgs.gov/of/1987/0112/report.pdf	Sanford, W. E., Pope, J. P., Selnick, D. L., Stumvoll, R. F. (2012). Simulation of groundwater flow in the shallow aquifer system of the Delmarva Peninsula, Maryland and Delaware. US Geological Survey Open-File Report 2012-1140, 68 pp. Accessed November 29, 2021 from https://pubs.usgs.gov/of/2012/1140/pdf/OFR_2012-1140.pdf	-	Approximated from Fig. 4 of Bachman et al. (1987) and Fig. 4 by Sanford et al. (2012)
Maryland Western Shores	North Atlantic Coastal Plain	Vroblesky, D.A., Fleck, W.B. (1991). Hydrogeologic Framework of the Coastal Plain of Maryland, Delaware, and the District of Columbia. U.S. Geological Survey Professional Paper 1404-E, 62 pp. Accessed April 1, 2021 from https://pubs.usgs.gov/pp/1404e/report.pdf	-	-	Approximated from Fig. 4 of Vroblesky and Fleck (1991).
North Carolina and Virginia Coastal Plain	North Atlantic Coastal Plain	Meng, A., Harsh, J.F. (1988). Hydrogeologic framework of the Virginia coastal plain. U.S. Geological Survey Professional Paper 1404-C, 85 pp. Accessed April 1, 2021 from https://pubs.usgs.gov/pp/pp1404-C/pdf/pp_1404-c.pdf	Winner Jr., M.D., Coble, R.W. (1989). Hydrogeologic framework of the North Carolina Coastal Plain aquifer system. U.S. Geological Survey Report 87-690, 167 pp. Accessed April 1, 2021 from https://pubs.usgs.gov/of/1987/0690/report.pdf	-	Approximated from Fig. 4 of Meng and Harsh (1988) and Fig. 4 of Winner Jr. and Coble (1989).
Long Island	-	Smolensky, D.A., Buxton, H.T., Shernoff, P.K. (1990). Hydrologic framework of Long Island, New York. U.S. Geological Survey Hydrologic Atlas 709, 3 plates. Accessed April 1, 2021 from https://pubs.usgs.gov/ha/709/plate-1.pdf	Olcott, P.G. (1995). Ground Water Atlas of the United States: Segment 12, Connecticut, Maine, Massachusetts, New Hampshire, New York, Rhode Island, Vermont. U.S. Geological Survey Hydrologic Atlas 730-M, 30 pp. Accessed April 14, 2021 from https://pubs.usgs.gov/ha/730m/report.pdf	Masterson, J. P., Pope, J. P., Fienen, M. N., Monti Jr, J., Nardi, M. R., Finkelstein, J. S. (2016). Assessment of groundwater availability in the Northern Atlantic Coastal Plain aquifer system from Long Island, New York, to North Carolina (No. 1829). US Geological Survey Professional Paper 1829, 90 pp. Accessed November 29, 2021 from https://pubs.usgs.gov/pp/1829/pp1829.pdf	Approximated from Plate 1 of Smolensky et al. (1990); Fig. 6 by Masterson et al. (2016); and Fig. 62 of Olcott (1995).
Moulton Valley	Interior Low Plateaus	Geological Survey of Alabama (2018). Assessment of groundwater resources in Alabama, 2010-16. Geological Survey of Alabama Bulletin 186, 462 pp. Accessed April 8, 2021 from https://www.gsa.state.al.us/img/Groundw	Miller, J.A. (1990). Ground Water Atlas of the United States: Segment 6, Alabama, Florida, Georgia, South Carolina. U.S. Geological Survey Hydrologic Atlas 730-G, 30 pp. Accessed April 5, 2021 from	-	Moulton Valley subarea approximated from Fig. 4 of Geological Survey of Alabama (2008) and Fig. 104 of Miller (1990). Broader Interior Low Plateaus outline approximated for Alabama from Fig. 4 of Geological

Aquifer System	Broader System	References	Delineation steps and sources	
Nashville Basin	Interior-Low Plateaus	ater/docs/assessment/00_B186_StatewideAssessment_Print_Document.pdf Miller, J.A. (1995). Ground Water Atlas of the United States: Segment 10, Illinois, Indiana, Kentucky, Ohio, Tennessee. U.S.-Geological Survey Hydrologic Atlas 730-K, 30 pp. Accessed April 11, 2021 from https://pubs.usgs.gov/ha/730k/report.pdf	https://www.nrc.gov/docs/ML1706/ML17060B027.pdf Brahana, J.V., Bradley, M.W. (1986a). Preliminary Delineation and Description of the Regional Aquifers of Tennessee: The Central Basin Aquifer System. U.S.-Geological Survey Water Resources Investigations Report 82-4002, 40 pp. Accessed April 11, 2021 from https://pubs.usgs.gov/wri/wri82-4002/pdf/wri82-4002_a.pdf Taylor, C.J., Nelson Jr., H.L. (2008). A compilation of provisional karst-geospatial data for the Interior-Low Plateaus physiographic region, central United States. U.S.-Geological Survey Data Series 339, 26 pp. Accessed April 11, 2021 from https://pubs.usgs.gov/ds/339/pdf/ds339_web.pdf	Survey of Alabama (2008) and Fig. 104 of Miller (1990). Broader Interior-Low Plateaus outline approximated for Tennessee, Illinois, Kentucky, Indiana from Fig. 78 of Miller (1990) and Fig. 1 of Taylor and Nelson (2008). "Central Basin" and "Nashville Basin" (seemingly similar areas) approximated from Fig. 1 of Taylor and Nelson (2008) and Fig. 1 of Brahana and Bradley (1986a)
Southern Highland Rim	Interior-Low Plateaus	Miller, J.A. (1995). Ground Water Atlas of the United States: Segment 10, Illinois, Indiana, Kentucky, Ohio, Tennessee. U.S.-Geological Survey Hydrologic Atlas 730-K, 30 pp. Accessed April 11, 2021 from https://pubs.usgs.gov/ha/730k/report.pdf	Brahana, J.V., Bradley, M.W. (1986b). Preliminary delineation and description of the regional aquifers of Tennessee—the Highland Rim Aquifer System. Water Resources Investigations Report 82-4054, 35 pp. Accessed April 11, 2021 from https://pubs.usgs.gov/wri/wri824054/pdf/wri82-4054_a.pdf Taylor, C.J., Nelson Jr., H.L. (2008). A compilation of provisional karst-geospatial data for the Interior-Low Plateaus physiographic region, central United States. U.S.-Geological Survey Data Series 339, 26 pp. Accessed April 11, 2021 from https://pubs.usgs.gov/ds/339/pdf/ds339_web.pdf	Broader Interior-Low Plateaus outline approximated for Tennessee, Illinois, Kentucky, Indiana from Fig. 78 of Miller (1990) and Fig. 1 of Taylor and Nelson (2008). Highland Rim subarea approximated from Fig. 1 of Taylor and Nelson (2008) and Fig. 1 of Brahana and Bradley (1986b)
Eastern Highland Rim	Interior-Low Plateaus	Miller, J.A. (1995). Ground Water Atlas of the United States: Segment 10, Illinois, Indiana, Kentucky, Ohio, Tennessee. U.S.-Geological Survey Hydrologic Atlas 730-K, 30 pp. Accessed April 11, 2021 from https://pubs.usgs.gov/ha/730k/report.pdf	Brahana, J.V., Bradley, M.W. (1986b). Preliminary delineation and description of the regional aquifers of Tennessee—the Highland Rim Aquifer System. Water Resources Investigations Report 82-4054, 35 pp. Accessed April 11, 2021 from https://pubs.usgs.gov/wri/wri824054/pdf/wri82-4054_a.pdf Taylor, C.J., Nelson Jr., H.L. (2008). A compilation of provisional karst-geospatial data for the Interior-Low Plateaus physiographic region, central United States. U.S.-Geological Survey Data Series 339, 26 pp. Accessed April 11, 2021 from https://pubs.usgs.gov/ds/339/pdf/ds339_web.pdf	Broader Interior-Low Plateaus outline approximated for Tennessee, Illinois, Kentucky, Indiana from Fig. 78 of Miller (1990) and Fig. 1 of Taylor and Nelson (2008). Highland Rim subarea approximated from Fig. 1 of Taylor and Nelson (2008) and Fig. 1 of Brahana and Bradley (1986b)
Western Highland Rim	Interior-Low Plateaus	Miller, J.A. (1995). Ground Water Atlas of the United States: Segment 10, Illinois, Indiana, Kentucky, Ohio, Tennessee. U.S.-Geological Survey Hydrologic Atlas 730-K, 30 pp. Accessed April 11, 2021 from https://pubs.usgs.gov/ha/730k/report.pdf	Brahana, J.V., Bradley, M.W. (1986b). Preliminary delineation and description of the regional aquifers of Tennessee—the Highland Rim Aquifer System. Water Resources Investigations Report 82-4054, 35 pp. Accessed April 11, 2021 from https://pubs.usgs.gov/wri/wri824054/pdf/wri82-4054_a.pdf Taylor, C.J., Nelson Jr., H.L. (2008). A compilation of provisional karst-geospatial data for the Interior-Low Plateaus physiographic region, central United States. U.S.-Geological Survey Data Series 339, 26 pp. Accessed April 11, 2021 from https://pubs.usgs.gov/ds/339/pdf/ds339_web.pdf	Broader Interior-Low Plateaus outline approximated for Tennessee, Illinois, Kentucky, Indiana from Fig. 78 of Miller (1990) and Fig. 1 of Taylor and Nelson (2008). Highland Rim subarea approximated from Fig. 1 of Taylor and Nelson (2008) and Fig. 1 of Brahana and Bradley (1986b). Northern limit approximated along the Cumberland River.

Aquifer System	Broader System	References	References	References	Delineation steps and sources
Western Pennyroyal Plateau	Interior Low Plateaus	Miller, J.A. (1995). Ground Water Atlas of the United States: Segment 10, Illinois, Indiana, Kentucky, Ohio, Tennessee. U.S. Geological Survey Hydrologic Atlas 730-K, 30 pp. Accessed April 11, 2021 from https://pubs.usgs.gov/ha/730k/report.pdf	Taylor, C.J., Nelson Jr., H.L. (2008). A compilation of provisional karst geospatial data for the Interior Low Plateaus physiographic region, central United States: U.S. Geological Survey Data Series 339, 26 pp. Accessed April 11, 2021 from https://pubs.usgs.gov/ds/339/pdf/ds339_web.pdf	Davidson, B. (2018). Kentucky-Interagency Groundwater Monitoring Network: Annual Report July 2017–June 2018. 28 pp. Accessed April 11, 2021 from http://www.uky.edu/KGS/water/gnet/itac17-18.pdf	Broader Interior Low Plateaus outline approximated for Tennessee, Illinois, Kentucky, Indiana from Fig. 78 of Miller (1990) and Fig. 1 of Taylor and Nelson (2008). Pennyroyal Plain approximated from Fig. 1 of Davidson (2018); divide between western and eastern portions approximated along Salt Lick Creek and Barren River
Eastern Pennyroyal Plateau	Interior Low Plateaus	Miller, J.A. (1995). Ground Water Atlas of the United States: Segment 10, Illinois, Indiana, Kentucky, Ohio, Tennessee. U.S. Geological Survey Hydrologic Atlas 730-K, 30 pp. Accessed April 11, 2021 from https://pubs.usgs.gov/ha/730k/report.pdf	Taylor, C.J., Nelson Jr., H.L. (2008). A compilation of provisional karst geospatial data for the Interior Low Plateaus physiographic region, central United States: U.S. Geological Survey Data Series 339, 26 pp. Accessed April 11, 2021 from https://pubs.usgs.gov/ds/339/pdf/ds339_web.pdf	Davidson, B. (2018). Kentucky-Interagency Groundwater Monitoring Network: Annual Report July 2017–June 2018. 28 pp. Accessed April 11, 2021 from http://www.uky.edu/KGS/water/gnet/itac17-18.pdf	Broader Interior Low Plateaus outline approximated for Tennessee, Illinois, Kentucky, Indiana from Fig. 78 of Miller (1990) and Fig. 1 of Taylor and Nelson (2008). Pennyroyal Plain approximated from Fig. 1 of Davidson (2018); northern limit approximated along the Salt River and Ohio River
Mitchell Plateau	Interior Low Plateaus	Miller, J.A. (1995). Ground Water Atlas of the United States: Segment 10, Illinois, Indiana, Kentucky, Ohio, Tennessee. U.S. Geological Survey Hydrologic Atlas 730-K, 30 pp. Accessed April 11, 2021 from https://pubs.usgs.gov/ha/730k/report.pdf	Taylor, C.J., Nelson Jr., H.L. (2008). A compilation of provisional karst geospatial data for the Interior Low Plateaus physiographic region, central United States: U.S. Geological Survey Data Series 339, 26 pp. Accessed April 11, 2021 from https://pubs.usgs.gov/ds/339/pdf/ds339_web.pdf	Florea, L. J., Hasenmueller, N. R., Branam, T. D., Frushour, S. S., Powell, R. L. (2018). Karst geology and hydrogeology of the Mitchell Plateau of south-central Indiana. In: Field Guide 54, Published by The Geological Society of America, pp. 95–112.	Broader Interior Low Plateaus outline approximated for Tennessee, Illinois, Kentucky, Indiana from Fig. 78 of Miller (1990) and Fig. 1 of Taylor and Nelson (2008). Mitchell Plain subarea approximated from Fig. 1 of Florea et al. (2008).
Shawnee Hills	Interior Low Plateaus	Miller, J.A. (1995). Ground Water Atlas of the United States: Segment 10, Illinois, Indiana, Kentucky, Ohio, Tennessee. U.S. Geological Survey Hydrologic Atlas 730-K, 30 pp. Accessed April 11, 2021 from https://pubs.usgs.gov/ha/730k/report.pdf	Taylor, C.J., Nelson Jr., H.L. (2008). A compilation of provisional karst geospatial data for the Interior Low Plateaus physiographic region, central United States: U.S. Geological Survey Data Series 339, 26 pp. Accessed April 11, 2021 from https://pubs.usgs.gov/ds/339/pdf/ds339_web.pdf	Leighton, M. M., Ekblaw, G. E., Horberg, L. (1948). Physiographic divisions of Illinois. The Journal of Geology, 56, 16–33.	Broader Interior Low Plateaus outline approximated for Tennessee, Illinois, Kentucky, Indiana from Fig. 78 of Miller (1990) and Fig. 1 of Taylor and Nelson (2008). Shawnee Hills approximated from Figs. 1 and 2 of Leighton et al. (1948)
Inner Bluegrass	Interior Low Plateaus	Miller, J.A. (1995). Ground Water Atlas of the United States: Segment 10, Illinois, Indiana, Kentucky, Ohio, Tennessee. U.S. Geological Survey Hydrologic Atlas 730-K, 30 pp. Accessed April 11, 2021 from https://pubs.usgs.gov/ha/730k/report.pdf	Taylor, C.J., Nelson Jr., H.L. (2008). A compilation of provisional karst geospatial data for the Interior Low Plateaus physiographic region, central United States: U.S. Geological Survey Data Series 339, 26 pp. Accessed April 11, 2021 from https://pubs.usgs.gov/ds/339/pdf/ds339_web.pdf	Scanlon, B. R. (1989). Physical Controls on Hydrochemical Variability in the Inner Bluegrass Karst Region of Central Kentucky. Groundwater, 27, 639–646.	Broader Interior Low Plateaus outline approximated for Tennessee, Illinois, Kentucky, Indiana from Fig. 78 of Miller (1990) and Fig. 1 of Taylor and Nelson (2008). Inner Bluegrass area approximated from Fig. 1 of Scanlon (1989).

Aquifer System	Broader System	References	Delineation steps and sources		
			https://pubs.usgs.gov/ds/339/pdf/ds339_web.pdf		
Western Outer Bluegrass	Interior Low Plateaus	Miller, J.A. (1995). Ground Water Atlas of the United States: Segment 10, Illinois, Indiana, Kentucky, Ohio, Tennessee. U.S. Geological Survey Hydrologic Atlas 730-K, 30 pp. Accessed April 11, 2021 from https://pubs.usgs.gov/ha/730k/report.pdf	Taylor, C.J., Nelson Jr., H.L. (2008). A compilation of provisional karst geospatial data for the Interior Low Plateaus physiographic region, central United States: U.S. Geological Survey Data Series 339, 26 pp. Accessed April 11, 2021 from https://pubs.usgs.gov/ds/339/pdf/ds339_web.pdf	Davidson, B. (2018). Kentucky Interagency Groundwater Monitoring Network: Annual Report July 2017–June 2018. 28 pp. Accessed April 11, 2021 from http://www.uky.edu/KGS/water/gnet/itac17-18.pdf	Broader Interior Low Plateaus outline approximated for Tennessee, Illinois, Kentucky, Indiana from Fig. 78 of Miller (1990) and Fig. 1 of Taylor and Nelson (2008). Outer Bluegrass approximated from Fig. 1 of Davidson (2018).
Eastern Outer Bluegrass	Interior Low Plateaus	Miller, J.A. (1995). Ground Water Atlas of the United States: Segment 10, Illinois, Indiana, Kentucky, Ohio, Tennessee. U.S. Geological Survey Hydrologic Atlas 730-K, 30 pp. Accessed April 11, 2021 from https://pubs.usgs.gov/ha/730k/report.pdf	Taylor, C.J., Nelson Jr., H.L. (2008). A compilation of provisional karst geospatial data for the Interior Low Plateaus physiographic region, central United States: U.S. Geological Survey Data Series 339, 26 pp. Accessed April 11, 2021 from https://pubs.usgs.gov/ds/339/pdf/ds339_web.pdf	Davidson, B. (2018). Kentucky Interagency Groundwater Monitoring Network: Annual Report July 2017–June 2018. 28 pp. Accessed April 11, 2021 from http://www.uky.edu/KGS/water/gnet/itac17-18.pdf	Broader Interior Low Plateaus outline approximated for Tennessee, Illinois, Kentucky, Indiana from Fig. 78 of Miller (1990) and Fig. 1 of Taylor and Nelson (2008). Outer Bluegrass approximated from Fig. 1 of Davidson (2018).
Crawford Upland	Interior Low Plateaus	Miller, J.A. (1995). Ground Water Atlas of the United States: Segment 10, Illinois, Indiana, Kentucky, Ohio, Tennessee. U.S. Geological Survey Hydrologic Atlas 730-K, 30 pp. Accessed April 11, 2021 from https://pubs.usgs.gov/ha/730k/report.pdf	Taylor, C.J., Nelson Jr., H.L. (2008). A compilation of provisional karst geospatial data for the Interior Low Plateaus physiographic region, central United States: U.S. Geological Survey Data Series 339, 26 pp. Accessed April 11, 2021 from https://pubs.usgs.gov/ds/339/pdf/ds339_web.pdf	Fenelon, J.M., Bobay, K.E., Greeman, T.K., Hoover, M.E., Cohen, D.A., Fowler, K.K., Woodfield, M.C., Durbin, J.M. (1994). Hydrogeologic atlas of aquifers in Indiana: U.S. Geological Survey Water-Resources Investigations Report 1992-4142, 197 pp. Accessed April 11, 2021 from https://pubs.er.usgs.gov/publication/wri924142	Broader Interior Low Plateaus outline approximated for Tennessee, Illinois, Kentucky, Indiana from Fig. 78 of Miller (1990) and Fig. 1 of Taylor and Nelson (2008). Crawford Upland approximated from Fig. 2 of Fenelon et al. (1994).
Norman Upland	Interior Low Plateaus	Miller, J.A. (1995). Ground Water Atlas of the United States: Segment 10, Illinois, Indiana, Kentucky, Ohio, Tennessee. U.S. Geological Survey Hydrologic Atlas 730-K, 30 pp. Accessed April 11, 2021 from https://pubs.usgs.gov/ha/730k/report.pdf	Taylor, C.J., Nelson Jr., H.L. (2008). A compilation of provisional karst geospatial data for the Interior Low Plateaus physiographic region, central United States: U.S. Geological Survey Data Series 339, 26 pp. Accessed April 11, 2021 from https://pubs.usgs.gov/ds/339/pdf/ds339_web.pdf	Fenelon, J.M., Bobay, K.E., Greeman, T.K., Hoover, M.E., Cohen, D.A., Fowler, K.K., Woodfield, M.C., Durbin, J.M. (1994). Hydrogeologic atlas of aquifers in Indiana: U.S. Geological Survey Water-Resources Investigations Report 1992-4142, 197 pp. Accessed April 11, 2021 from https://pubs.er.usgs.gov/publication/wri924142	Broader Interior Low Plateaus outline approximated for Tennessee, Illinois, Kentucky, Indiana from Fig. 78 of Miller (1990) and Fig. 1 of Taylor and Nelson (2008). Norman Upland approximated from Fig. 2 of Fenelon et al. (1994).
Muscatatuck Plateau and Dearborn Upland	-	Gray, H.H. (2001). Map of Indiana showing physiographic divisions. Indiana Geological Survey Miscellaneous Map 69, 15 pp. Accessed April 12, 2021 from	Schrader, G.P. (2004). Unconsolidated Aquifer Systems of Ripley County, Indiana. Indiana Department of Natural Resources,	-	Approximated from map by Gray (2001). Muscatatuck Plateau and Dearborn Upland combined as local hydrogeologic report refers to

Aquifer System	Broader System	References	Delineation steps and sources
		https://igws.indiana.edu/ReferenceDocs/Maps/PhysiographicRegions.pdf	Division of Water Report, 3 pp. Accessed April 12, 2021 from https://www.in.gov/dnr/water/files/ripl/ey_unconsolidated_text.pdf
Newcastle Till Plain	Central-Lowland Till Plain	Gray, H.H. (2001). Map of Indiana showing physiographic divisions. Indiana Geological Survey Miscellaneous Map 69, 15 pp. Accessed April 12, 2021 from https://igws.indiana.edu/ReferenceDocs/Maps/PhysiographicRegions.pdf	Fenneman, N.M., Johnson, D.W. (1946). Physiographic divisions of the conterminous United States. U.S. Geological Survey map, 1:7,000,000 Scale.
Bluffton Till Plain	Central-Lowland Till Plain	Gray, H.H. (2001). Map of Indiana showing physiographic divisions. Indiana Geological Survey Miscellaneous Map 69, 15 pp. Accessed April 12, 2021 from https://igws.indiana.edu/ReferenceDocs/Maps/PhysiographicRegions.pdf	Brockman, S. (1998). Physiographic regions of Ohio. Ohio Department of Natural Resources, Division of Geological Survey map. Accessed April 12, 2021 from https://www.epa.state.oh.us/portals/27/SIP/Nonattain/F2-physiographic_regions_of_Ohio.pdf
Columbus Lowland and Darby Plain	Central-Lowland Till Plain	Brockman, S. (1998). Physiographic regions of Ohio. Ohio Department of Natural Resources, Division of Geological Survey map. Accessed April 12, 2021 from https://www.epa.state.oh.us/portals/27/SIP/Nonattain/F2-physiographic_regions_of_Ohio.pdf	Fenneman, N.M., Johnson, D.W. (1946). Physiographic divisions of the conterminous United States. U.S. Geological Survey map, 1:7,000,000 Scale.
Bellefontaine Upland and Mad River Interlobate Plain	Central-Lowland Till Plain	Brockman, S. (1998). Physiographic regions of Ohio. Ohio Department of Natural Resources, Division of Geological Survey map. Accessed April 12, 2021 from https://www.epa.state.oh.us/portals/27/SIP/Nonattain/F2-physiographic_regions_of_Ohio.pdf	Fenneman, N.M., Johnson, D.W. (1946). Physiographic divisions of the conterminous United States. U.S. Geological Survey map, 1:7,000,000 Scale.
Galion Glaciated Low Plateau	Central-Lowland Till Plain	Brockman, S. (1998). Physiographic regions of Ohio. Ohio Department of Natural Resources, Division of Geological Survey map. Accessed April 12, 2021 from https://www.epa.state.oh.us/portals/27/SIP/Nonattain/F2-physiographic_regions_of_Ohio.pdf	Fenneman, N.M., Johnson, D.W. (1946). Physiographic divisions of the conterminous United States. U.S. Geological Survey map, 1:7,000,000 Scale.
			these two regions as a single unit (e.g., quote Schrader (2004): "Muscatatuck Plateau / Dearborn Upland Till Aquifer Subsystem.")
			Broader Till Plain physiographic region approximated from Fenneman and Johnson (1946). Newcastle Till Plain approximated for Indiana from map by Gray (2001) and for Ohio following the approximate border of the "Southern Ohio Loamy Till Plain" defined in map by Brockman (1998).
			Broader Till Plain physiographic region approximated from Fenneman and Johnson (1946). Bluffton Till Plain approximated for Indiana from map by Gray (2001) and for Ohio following the approximate border of the "Central Ohio Clayey Till Plain" defined in map by Brockman (1998).
			Broader Till Plain physiographic region approximated from Fenneman and Johnson (1946). Columbus Lowland and Darby Plain areas combined with outlines approximated from the map by Brockman (1998).
			Broader Till Plain physiographic region approximated from Fenneman and Johnson (1946). Bellefontaine Upland and Mad River Interlobate Plain areas combined with outlines approximated from the map by Brockman (1998).
			Broader Till Plain physiographic region approximated from Fenneman and Johnson (1946). Galion Glaciated Low Plateau approximated from the map by Brockman (1998).

Aquifer System	Broader System	References			Delineation steps and sources
Illinoian Till Plain	Central-Lowland Till Plain	Brockman, S. (1998). Physiographic regions of Ohio. Ohio Department of Natural Resources, Division of Geological Survey map. Accessed April 12, 2024 from https://www.epa.state.oh.us/portals/27/SIP/Nonattain/F2-physiographic_regions_of_Ohio.pdf	Fenneman, N.M., Johnson, D.W. (1946). Physiographic divisions of the conterminous United States. U.S. Geological Survey map, 1:7,000,000 Scale.	-	Broader Till Plain physiographic region approximated from Fenneman and Johnson (1946). Illinoian Till Plain approximated from the map by Brockman (1998).
Tipton Till Plain	Central-Lowland Till Plain	Gray, H.H. (2001). Map of Indiana showing physiographic divisions. Indiana Geological Survey Miscellaneous Map 69, 15 pp. Accessed April 12, 2021 from https://igws.indiana.edu/ReferenceDocs/Maps/PhysiographicRegions.pdf	Fenneman, N.M., Johnson, D.W. (1946). Physiographic divisions of the conterminous United States. U.S. Geological Survey map, 1:7,000,000 Scale.	-	Broader Till Plain physiographic region approximated from Fenneman and Johnson (1946). Tipton Till Plain approximated from the map by Gray (2001).
Mount Vernon Hill County	Central-Lowland Till Plain	Leighton, M. M., Ekblaw, G. E., Horberg, L. (1948). Physiographic divisions of Illinois. The Journal of Geology, 56, 16-33.	Fenneman, N.M., Johnson, D.W. (1946). Physiographic divisions of the conterminous United States. U.S. Geological Survey map, 1:7,000,000 Scale.	-	Broader Till Plain physiographic region approximated from Fenneman and Johnson (1946). Mount Vernon Hill County approximated from Figs. 1 and 2 of Leighton et al. (1948)
Springfield Plain	Central-Lowland Till Plain	Leighton, M. M., Ekblaw, G. E., Horberg, L. (1948). Physiographic divisions of Illinois. The Journal of Geology, 56, 16-33.	Fenneman, N.M., Johnson, D.W. (1946). Physiographic divisions of the conterminous United States. U.S. Geological Survey map, 1:7,000,000 Scale.	Gray, H.H. (2001). Map of Indiana showing physiographic divisions. Indiana Geological Survey Miscellaneous Map 69, 15 pp. Accessed April 12, 2021 from https://igws.indiana.edu/ReferenceDocs/Maps/PhysiographicRegions.pdf	Broader Till Plain physiographic region approximated from Fenneman and Johnson (1946). Springfield Plain approximated from Figs. 1 and 2 of Leighton et al. (1948); northern portion of "Wabash Lowland" delineated by Gray (2001) was included for the Indiana portion of the subarea
Central Wabash and Bloomington Ridged Plain	Central-Lowland Till Plain	Fenneman, N.M., Johnson, D.W. (1946). Physiographic divisions of the conterminous United States. U.S. Geological Survey map, 1:7,000,000 Scale.	Leighton, M. M., Ekblaw, G. E., Horberg, L. (1948). Physiographic divisions of Illinois. The Journal of Geology, 56, 16-33.	Gray, H.H. (2001). Map of Indiana showing physiographic divisions. Indiana Geological Survey Miscellaneous Map 69, 15 pp. Accessed April 12, 2021 from https://igws.indiana.edu/ReferenceDocs/Maps/PhysiographicRegions.pdf	Broader Till Plain physiographic region approximated from Fenneman and Johnson (1946). Bloomington Ridged Plain approximated from Figs. 1 and 2 of Leighton et al. (1948); "Central Wabash Valley" delineated by Gray (2001) was included for the Indiana portion of the subarea
Iroquois Till Plains	Central-Lowland Till Plain	Fenneman, N.M., Johnson, D.W. (1946). Physiographic divisions of the conterminous United States. U.S. Geological Survey map, 1:7,000,000 Scale.	Gray, H.H. (2001). Map of Indiana showing physiographic divisions. Indiana Geological Survey Miscellaneous Map 69, 15 pp. Accessed April 12, 2021 from https://igws.indiana.edu/ReferenceDocs/Maps/PhysiographicRegions.pdf	-	Broader Till Plain physiographic region approximated from Fenneman and Johnson (1946). "Iroquois Till Plain" delineated from Gray (2001).
Michigan Basin	-	Olcott, P.G. (1992). Groundwater Atlas of the United States: Segment 9 Iowa, Michigan, Minnesota, Wisconsin. U.S.	Westjohn, D.B., Weaver, T.L. (1998). Hydrogeologic framework of the Michigan Basin regional aquifer	-	Approximated from Figs. 50-64 by Olcott (1992) and Fig. 1 by Westjohn and Weaver (1998)

Aquifer System	Broader System	References			Delineation steps and sources
		Geological Survey Hydrologic Atlas 730-J, 33 pp. Accessed April 12, 2021 from https://pubs.usgs.gov/hal/730j/report.pdf	system. US Geological Survey Professional Paper 1418, 55 pp. Accessed November 29, 2021 from https://pubs.usgs.gov/pp/1418/report.pdf		
Silurian-Devonian Aquifers in Northern Michigan	-	Olcott, P.G. (1992). Groundwater Atlas of the United States: Segment 9 Iowa, Michigan, Minnesota, Wisconsin. U.S. Geological Survey Hydrologic Atlas 730-J, 33 pp. Accessed April 12, 2021 from https://pubs.usgs.gov/hal/730j/report.pdf	-	-	Approximated from Fig. 79 by Olcott (1992).
Wisconsin Precambrian Aquifer	-	U.S. Geological Survey (1984). National water summary 1984: Hydrologic events, selected water quality trends, and ground-water resources. U.S. Geological Survey Water Supply Paper 2275, 477 pp. Accessed April 13, 2021 from https://pubs.usgs.gov/wsp/2275/report.pdf	-	-	Approximated from Fig. 1 of the "National Water Summary: Wisconsin" (page 449) by the U.S. Geological Survey (1984).
Western Kankakee Plains	-	Fenneman, N.M., Johnson, D.W. (1946). Physiographic divisions of the conterminous United States. U.S. Geological Survey map, 1:7,000,000 Scale.	Leighton, M. M., Ekblaw, G. E., Horberg, L. (1948). Physiographic divisions of Illinois. The Journal of Geology, 56, 16-33.	-	Broader Till Plain physiographic region approximated from Fenneman and Johnson (1946). Kankakee Plain in Illinois approximated from Figs. 1 and 2 of Leighton et al. (1948). Eastern margin of subarea approximated along the Iroquois River.
Eastern Kankakee Plains	-	Fenneman, N.M., Johnson, D.W. (1946). Physiographic divisions of the conterminous United States. U.S. Geological Survey map, 1:7,000,000 Scale.	Leighton, M. M., Ekblaw, G. E., Horberg, L. (1948). Physiographic divisions of Illinois. The Journal of Geology, 56, 16-33.	Gray, H.H. (2001). Map of Indiana showing physiographic divisions. Indiana Geological Survey Miscellaneous Map 69, 15 pp. Accessed April 12, 2021 from https://igws.indiana.edu/ReferenceDocs/Maps/PhysiographicRegions.pdf	Broader Till Plain physiographic region approximated from Fenneman and Johnson (1946). Kankakee Plain in Illinois approximated from Figs. 1 and 2 of Leighton et al. (1948). Indiana portion of subarea approximated from Gray (2001). Western margin of subarea approximated at northwest margin of the Iroquois Till Plains (located to the southwest of this subarea)
Eastern Silurian-Devonian Aquifers	Northern Midwest Aquifer System	Young, H.L. (1992). Hydrogeology of the Cambrian-Ordovician Aquifer System in the Northern Midwest, United States. U.S. Geological Survey Professional Paper 1405-B, 108 pp. Accessed April 12, 2021 from https://pubs.usgs.gov/pp/1405b/report.pdf	Wilson, J.T. (2012). Water quality assessment of the Cambrian-Ordovician aquifer system in the northern Midwest, United States. U.S. Geological Survey Scientific Investigations Report 2011-5229, 174 pp. Accessed April 12, 2021 from	Kay, R. T., Krasko, K. A. (1996). Ground Water Levels in Aquifers Used for Residential Supply, Campton Township, Kane County, Illinois. U.S. Geological Survey Water Resources Investigations Report 96-4009, Accessed April 12,	Broader aquifer system title and its location approximated from Fig. 1 of Young (1992). Silurian and Devonian-aged rock subcrop approximated from Fig. 9 of Wilson (2012). These Silurian-Devonian aquifers subcrop along the western side of Lake Michigan; however, they are underlain by the

Aquifer System	Broader System	References		Delineation steps and sources	
			https://pubs.usgs.gov/sir/2011/5229/pdf/SIR20115229_web.pdf	2021 from https://pubs.usgs.gov/wri/1996/4009/report.pdf	Cambrian-Ordovician aquifers that subcrop to the west (see Fig. 11 of Young (1992)) and thus this "Silurian-Devonian" area is also included as part of the broader "Cambrian-Ordovician Aquifer System". Further, wells in this area (i.e., the "Silurian-Devonian Area") are known to penetrate the deep "Mount Simon Sandstone" aquifer that is Cambrian in age (for another example, a 711-foot well penetrated the Galena-Platteville Formation in 1860 see page B67 of Young (1992)).
Western Cambrian-Ordovician Aquifers	Northern Midwest Aquifer System	Young, H.L. (1992). Hydrogeology of the Cambrian-Ordovician Aquifer System in the Northern Midwest, United States. U.S. Geological Survey Professional Paper 1405-B, 108 pp. Accessed April 12, 2021 from https://pubs.usgs.gov/pp/1405b/report.pdf	Wilson, J.T. (2012). Water quality assessment of the Cambrian-Ordovician aquifer system in the northern Midwest, United States. U.S. Geological Survey Scientific Investigations Report 2011-5229, 174 pp. Accessed April 12, 2021 from https://pubs.usgs.gov/sir/2011/5229/pdf/SIR20115229_web.pdf	-	Broader aquifer system title and its location approximated from Fig. 1 of Young (1992). Subcrop geology approximated from Fig. 9 of Wilson (2012). Southern margin extended south of the subcrop margin on the basis of the presence of deep wells.
Eastern Cambrian-Ordovician Aquifers	Northern Midwest Aquifer System	Young, H.L. (1992). Hydrogeology of the Cambrian-Ordovician Aquifer System in the Northern Midwest, United States. U.S. Geological Survey Professional Paper 1405-B, 108 pp. Accessed April 12, 2021 from https://pubs.usgs.gov/pp/1405b/report.pdf	Wilson, J.T. (2012). Water quality assessment of the Cambrian-Ordovician aquifer system in the northern Midwest, United States. U.S. Geological Survey Scientific Investigations Report 2011-5229, 174 pp. Accessed April 12, 2021 from https://pubs.usgs.gov/sir/2011/5229/pdf/SIR20115229_web.pdf	-	Broader aquifer system title and its location approximated from Fig. 1 of Young (1992). Subcrop geology approximated from Fig. 9 of Wilson (2012). Southern margin extended south of the subcrop margin on the basis of the presence of deep wells.
Northern Cambrian-Ordovician Aquifers	Northern Midwest Aquifer System	Young, H.L. (1992). Hydrogeology of the Cambrian-Ordovician Aquifer System in the Northern Midwest, United States. U.S. Geological Survey Professional Paper 1405-B, 108 pp. Accessed April 12, 2021 from https://pubs.usgs.gov/pp/1405b/report.pdf	Wilson, J.T. (2012). Water quality assessment of the Cambrian-Ordovician aquifer system in the northern Midwest, United States. U.S. Geological Survey Scientific Investigations Report 2011-5229, 174 pp. Accessed April 12, 2021 from https://pubs.usgs.gov/sir/2011/5229/pdf/SIR20115229_web.pdf	-	Broader aquifer system title and its location approximated from Fig. 1 of Young (1992). Subcrop geology approximated from Fig. 9 of Wilson (2012).
Upper Carbonate Aquifer	Northern Midwest Aquifer System	Olcott, P.G. (1992). Groundwater Atlas of the United States: Segment 9 Iowa, Michigan, Minnesota, Wisconsin. U.S.	Young, H.L. (1992). Hydrogeology of the Cambrian-Ordovician Aquifer System in the Northern Midwest,	-	Broader aquifer system title and its location approximated from Fig. 1 of Young (1992). Upper Carbonate

Aquifer System	Broader System	References		Delineation steps and sources
		Geological Survey Hydrologic Atlas 730-J, 33 pp. Accessed April 12, 2021 from https://pubs.usgs.gov/ha/730j/report.pdf	United States. U.S. Geological Survey Professional Paper 1405-B, 108 pp. Accessed April 12, 2021 from https://pubs.usgs.gov/pp/1405b/report.pdf	Aquifer location approximated from Fig. 17 of Olcott (1992).
Mississippian-Silurian-Devonian Carbonates	Northern Midwest Aquifer System	Olcott, P.G. (1992). Groundwater Atlas of the United States: Segment 9 Iowa, Michigan, Minnesota, Wisconsin. U.S. Geological Survey Hydrologic Atlas 730-J, 33 pp. Accessed April 12, 2021 from https://pubs.usgs.gov/ha/730j/report.pdf	Young, H.L. (1992). Hydrogeology of the Cambrian-Ordovician Aquifer System in the Northern Midwest, United States. U.S. Geological Survey Professional Paper 1405-B, 108 pp. Accessed April 12, 2021 from https://pubs.usgs.gov/pp/1405b/report.pdf	Broader aquifer system title and its location approximated from Fig. 1 of Young (1992). Subarea location approximated from Figs. 10 and 14 of Olcott (1992).
Upper Peninsula Jacobeville Sandstone and Cambrian-Ordovician and Silurian-Devonian Aquifers	Northern Midwest Aquifer System	Olcott, P.G. (1992). Groundwater Atlas of the United States: Segment 9 Iowa, Michigan, Minnesota, Wisconsin. U.S. Geological Survey Hydrologic Atlas 730-J, 33 pp. Accessed April 12, 2021 from https://pubs.usgs.gov/ha/730j/report.pdf	-	Approximated from Figs. 77, 97 and 133 by Olcott (1992).
Northeast Missouri Carbonates	Northern Midwest Aquifer System	Miller, J.A. (1995). Ground Water Atlas of the United States: Segment 10, Illinois, Indiana, Kentucky, Ohio, Tennessee. U.S. Geological Survey Hydrologic Atlas 730-K, 30 pp. Accessed April 11, 2021 from https://pubs.usgs.gov/ha/730k/report.pdf	Miller, J.A., Appel, C.L. (1997). Ground Water Atlas of the United States: Segment 3, Kansas, Missouri, Nebraska. U.S. Geological Survey Hydrologic Atlas 730-D, 26 pp. Accessed April 13, 2021 from https://pubs.usgs.gov/ha/730d/report.pdf	Approximated from Fig. 9 of Miller and Appel (1997) and Fig. 12 of Miller (1995).
Central Cumberland Plateau and Sequatchie Valley	Appalachian Plateaus	Brahana, J.V., Macy, J.A., Mulderink, D., Zemo, D. (1986). Preliminary delineation and description of the regional aquifers of Tennessee-Cumberland plateau aquifer system. U.S. Geological Survey, Water Resources Investigations Report 82-338, 29 pp. Accessed April 13, 2021 from https://pubs.usgs.gov/wri/wri82-338/pdf/wri82-338_a.pdf	-	Approximated from Fig. 2 by Brahana et al. (1986)
Southern Cumberland Plateau-Pottsville Aquifer	Appalachian Plateaus	Jennings, S.P. (2013). Hydrogeology and groundwater assessment of the water distribution area of the town of Hodges water department, Franklin and Marion Counties, Alabama. Geological Survey of Alabama Report, 15pp. Accessed April 8, 2021 from https://www.ogb.state.al.us/img/Groundwater/OFR/OFR1311.pdf	Geological Survey of Alabama (2018). Assessment of groundwater resources in Alabama, 2010-16. Geological Survey of Alabama Bulletin 486, 462 pp. Accessed April 8, 2021 from https://www.gsa.state.al.us/img/Groundwater/docs/assessment/00_B186_StatewideAssessment_Print_Document.pdf	Physiographic provinces from Fig. 4 of Tew (2013) and Fig. 4 of Geological Survey of Alabama (2018). Aquifer recharge areas defined in Fig. 6 of Geological Survey of Alabama (2018). Northern limit of area was approximated at Cuntersville Lake.

Aquifer System	Broader System	References			Delineation steps and sources
Northern Cumberland Plateau and Mountains	Appalachian Plateaus	Brahana, J.V., Macy, J.A., Mulderink, D., Zemo, D. (1986). Preliminary delineation and description of the regional aquifers of Tennessee-Cumberland plateau aquifer system. U.S. Geological Survey, Water-Resources Investigations Report 82-338, 29 pp. Accessed April 13, 2021 from https://pubs.usgs.gov/wri/wri82-338/pdf/wrir_82-338_a.pdf	-	-	Approximated from Fig. 2 by Brahana et al. (1986). Broader Appalachian Plateau Aquifer System approximated from Fig. 3 of Trapp Jr. and Horn (1997)
Southwestern Allegheny Plateau	Appalachian Plateaus	Brockman, S. (1998). Physiographic regions of Ohio. Ohio Department of Natural Resources, Division of Geological Survey map. Accessed April 12, 2021 from https://www.epa.state.oh.us/portals/27/ISI/P/Nonattain/F2-physiographic_regions_of_Ohio.pdf	-	-	Approximated from map by Brockman (1998). Northern boundary approximated along the Hackman River.
Western Allegheny Plateau	Appalachian Plateaus	Brockman, S. (1998). Physiographic regions of Ohio. Ohio Department of Natural Resources, Division of Geological Survey map. Accessed April 12, 2021 from https://www.epa.state.oh.us/portals/27/ISI/P/Nonattain/F2-physiographic_regions_of_Ohio.pdf	-	-	Approximated from map by Brockman (1998). Southern boundary approximated along the Hackman River. Broader Appalachian Plateau Aquifer System approximated from Fig. 3 of Trapp Jr. and Horn (1997)
Southeastern Allegheny Plateau	Appalachian Plateaus	Trapp Jr., H., Horn, M.A. (1997). Ground water atlas of the United States: Segment 11. U.S. Geological Survey Hydrologic Investigations Atlas 730-L, 26 pp. Accessed April 13, 2021 from https://pubs.usgs.gov/ha/730l/report.pdf	-	-	Broader Appalachian Plateau Aquifer System approximated from Fig. 3 of Trapp Jr. and Horn (1997)
Central Allegheny Plateau	Appalachian Plateaus	Trapp Jr., H., Horn, M.A. (1997). Ground water atlas of the United States: Segment 11. U.S. Geological Survey Hydrologic Investigations Atlas 730-L, 26 pp. Accessed April 13, 2021 from https://pubs.usgs.gov/ha/730l/report.pdf	-	-	Broader Appalachian Plateau Aquifer System approximated from Fig. 3 of Trapp Jr. and Horn (1997)
Northern Allegheny Plateau	Appalachian Plateaus	Trapp Jr., H., Horn, M.A. (1997). Ground water atlas of the United States: Segment 11. U.S. Geological Survey Hydrologic Investigations Atlas 730-L, 26 pp. Accessed April 13, 2021 from https://pubs.usgs.gov/ha/730l/report.pdf	-	-	Broader Appalachian Plateau Aquifer System approximated from Fig. 3 of Trapp Jr. and Horn (1997)
Southern Valley and Ridge	Valley and Ridge Aquifer System	Johnson, G.C., Zimmerman, T.M., Lindsey, B.D., Gross, E.L. (2011). Factors affecting groundwater quality in the Valley and Ridge aquifers, eastern United States, 1993–2002. U.S. Geological Survey of Alabama (2018). Assessment of groundwater resources in Alabama, 2010–16. Geological Survey of Alabama Bulletin 186, 462 pp. Accessed April	-	-	Northern margin approximated along the Watauga River. Broader Valley and Ridge Aquifer System outline approximated from Fig. 4 of Johnson et al. (2011), with

Aquifer-System	Broader-System	References			Delineation-steps-and-sources
		Geological Survey Scientific Investigations Report 2011-5115, 84 pp. Accessed April 13, 2021 from https://pubs.usgs.gov/sir/2011/5115/supp/ort/sir2011-5115.pdf	8, 2021 from https://www.gea.state.al.us/img/groundwater/docs/assessment/00_B186_StatewideAssessment_Print_Document.pdf		southernmost extent approximated from Fig. 4 of Geological Survey of Alabama (2018)
Southcentral Valley and Ridge	Valley and Ridge Aquifer-System	Johnson, G.C., Zimmerman, T.M., Lindsey, B.D., Gross, E.L. (2011). Factors affecting groundwater quality in the Valley and Ridge aquifers, eastern United States, 1993-2002. U.S. Geological Survey Scientific Investigations Report 2011-5115, 84 pp. Accessed April 13, 2021 from https://pubs.usgs.gov/sir/2011/5115/supp/ort/sir2011-5115.pdf	Brahana, J.V., Macy, J.A., Mulderink, D., Zemo, D. (1986). Preliminary delineation and description of the regional aquifers of Tennessee-Cumberland plateau aquifer system. U.S. Geological Survey, Water Resources Investigations Report 82-338, 29 pp. Accessed April 13, 2021 from https://pubs.usgs.gov/wri/wrir82-338/pdf/wrir_82-338-a.pdf	-	Northern margin approximated along the Hiwassee River. Broader Valley and Ridge Aquifer-System outline approximated from Fig. 1 of Johnson et al. (2011) and the Tennessee portion approximated from Fig. 1 of Brahana et al. (1986)
Northcentral Valley and Ridge	Valley and Ridge Aquifer-System	Johnson, G.C., Zimmerman, T.M., Lindsey, B.D., Gross, E.L. (2011). Factors affecting groundwater quality in the Valley and Ridge aquifers, eastern United States, 1993-2002. U.S. Geological Survey Scientific Investigations Report 2011-5115, 84 pp. Accessed April 13, 2021 from https://pubs.usgs.gov/sir/2011/5115/supp/ort/sir2011-5115.pdf	-	-	Broader Valley and Ridge Aquifer System outline approximated from Fig. 1 of Johnson et al. (2011). Southern margin approximated at Deep Creek Lake and North Fork Shenandoah River.
Northern Valley and Ridge	Valley and Ridge Aquifer-System	Johnson, G.C., Zimmerman, T.M., Lindsey, B.D., Gross, E.L. (2011). Factors affecting groundwater quality in the Valley and Ridge aquifers, eastern United States, 1993-2002. U.S. Geological Survey Scientific Investigations Report 2011-5115, 84 pp. Accessed April 13, 2021 from https://pubs.usgs.gov/sir/2011/5115/supp/ort/sir2011-5115.pdf	-	-	Southern boundary approximated at Deep Creek Lake and North Fork Shenandoah River. Broader Valley and Ridge Aquifer-System outline approximated from Fig. 1 of Johnson et al. (2011).
Southern St. Lawrence Lowlands	St. Lawrence Lowlands	Parent, M., Rivard, C., Lefebvre, R., Carrier, M.-A., Séjourné, S. (2014). Hydrogeological systems of the Montérégie Est region, southern Québec: Fieldtrip Guidebook, GeoMontreal 2013 Conference, Geological Survey of Canada Open File 7605, 43 pp.	Olcott, P.G. (1995). Ground Water Atlas of the United States: Segment 12, Connecticut, Maine, Massachusetts, New Hampshire, New York, Rhode Island, Vermont. U.S. Geological Survey Hydrologic Atlas 730-M, 30 pp. Accessed April 14, 2021 from https://pubs.usgs.gov/ha/730m/report.pdf	Patenaude, M., Baudron, P., Labelle, L., Masse-Dufresne, J. (2020). Evaluating bank-filtration occurrence in the Province of Quebec (Canada) with a GIS approach. Water 12, 662.	St. Lawrence Lowlands hydrogeological context approximated from Fig. 2 of Parent et al. (2014), including the divide between northern and southern portions of the broader aquifer system) and Fig. 4 by Patenaude et al. (2020); see also Fig. 102 by Olcott (1995).
Northern St. Lawrence Lowlands	St. Lawrence Lowlands	Parent, M., Rivard, C., Lefebvre, R., Carrier, M.-A., Séjourné, S. (2014).	Olcott, P.G. (1995). Ground Water Atlas of the United States: Segment	Patenaude, M., Baudron, P., Labelle, L., Masse-Dufresne,	St. Lawrence Lowlands hydrogeological context

Aquifer System	Broader System	References	Delineation steps and sources
		Hydrogeological systems of the Montérégie-Est region, southern Québec: Fieldtrip Guidebook, GeoMontreal 2013 Conference, Geological Survey of Canada Open File 7605, 43 pp.	J. (2020). Evaluating bank-filtration occurrence in the Province of Quebec (Canada) with a GIS approach. Water 12, 662.
Cape Cod Aquifer	-	Frimpter, M.H., Gay, F.B. (1979). Chemical quality of ground water on Cape Cod, Massachusetts. U.S. Geological Survey Water Resources Investigations Report 79-65, 20 pp. Accessed April 14, 2021 from https://pubs.usgs.gov/wri/1979/0065/report.pdf	Masterson, J.P., Walter, D.A. (2009). Hydrogeology and groundwater resources of the coastal aquifers of southeastern Massachusetts. U.S. Geological Survey Circular 1338, 16 pp. Accessed April 14, 2021 from https://pubs.usgs.gov/circ/circ1338/pdf/circular%202009-1338_508.pdf
Plymouth-Carver-Kingston-Duxbury Aquifer System	-	Masterson, J.P., Walter, D.A. (2009). Hydrogeology and groundwater resources of the coastal aquifers of southeastern Massachusetts. U.S. Geological Survey Circular 1338, 16 pp. Accessed April 14, 2021 from https://pubs.usgs.gov/circ/circ1338/pdf/circular%202009-1338_508.pdf	-
Northern San Juan Basin	-	Kernodle, J. M. (1996). Hydrogeology and steady-state simulation of groundwater flow in the San Juan Basin, New Mexico, Colorado, Arizona, and Utah. U.S. Geological Survey Water Resources Investigations Report 95-4187, 126 pp. Accessed March 12, 2021 from https://pubs.usgs.gov/wri/1995/4187/report.pdf	Welder, G. E. (1986). Plan of study for the regional aquifer system analysis of the San Juan structural basin, New Mexico, Colorado, Arizona, and Utah. U.S. Geological Survey Water Resources Investigations Report 85-4204, 27 pp. Accessed April 14, 2021 from https://pubs.usgs.gov/wri/1985/4204/report.pdf
Central Minnesota Surficial and Buried Sand and Gravel Aquifers	-	Minnesota Department of Natural Resources (2021). Minnesota Groundwater Provinces 2021. Minnesota Department of Natural Resources map-2 pp. Accessed April 14, 2021 from https://files.dnr.state.mn.us/waters/groundwater_section/provinces/2021-provinces.pdf	-
Sardis-Vedder Aquifer	-	City of Chilliwack (2021). Groundwater Protection. Webpage accessed March 12, 2021 from https://www.chilliwack.com/main/page.cfm?id=205	-

Aquifer System	Broader System	References			Delineation steps and sources
Eastern Dakota Aquifer	-	Leonard, R. B., Signor, D. C., Jorgensen, D. G., Helgesen, J. O. (1983). Geohydrology and hydrochemistry of the Dakota Aquifer, central United States. Journal of the American Water Resources Association, 19(6), 903-912.	Prior, J. C., Boekhoff, J. L., Howes, M. R., Libra, R. D., VanDorpe, P. E. (2003). Iowa's Groundwater Basics. Iowa Department of Natural Resources Report, 92 pp. Accessed November 29, 2021 from https://e-iihr34.iihr.uiowa.edu/publications/uploads/2014-08-24_08-08-24_es-06.pdf	-	Approximated from Fig. 9 of Leonard et al. (1983); page 30 of Prior et al. (2003)
Darton's Dakota Aquifer	-	Leonard, R. B., Signor, D. C., Jorgensen, D. G., Helgesen, J. O. (1983). Geohydrology and hydrochemistry of the Dakota Aquifer, central United States. Journal of the American Water Resources Association, 19(6), 903-912.	Bredehoeft, J. D., Neuzil, C. E., Milly, P. C. D. (1983). Regional flow in the Dakota aquifer: A study of the role of confining layers. US Geological Survey Water Supply Paper 2237, 50 pp. Accessed on February 9, 2021 from https://pubs.er.usgs.gov/publication/wsp2237	Macfarlane, P. A., Doveton, J. H., Whittlemore, D. O. (1998). User's Guide to the Dakota Aquifer in Kansas. Kansas Geological Survey. Accessed February 9, 2021 from http://www.kgs.ku.edu/Publications/Bulletins/TS2/index.html	Approximated the complete boundary of the Dakota Formation from Fig. 6 of Leonard et al. (1983). Other portions of the Dakota Aquifer were digitized from page 30 of Prior et al. (2003; Iowa portion) and Macfarlane et al. (1998; for eastern margin of Kansas' Dakota Aquifer (Fig. 2 of Macfarlane et al., 1998) and general boundaries extending into Canada (Fig. 1 of Macfarlane et al., 1998)). This specific portion of the Dakota was delineated by exploring well completion depths, mainly drawing from well completion records for South Dakota (specifically, records of deep wells).
Great Plains Dakota Aquifer	-	Whittlemore, D. O., Macfarlane, P. A., Wilson, B. B. (2014). Water Resources of the Dakota Aquifer in Kansas. Kansas Geological Survey Bulletin 260, 68 pp. Accessed April 15, 2021 from http://www.kgs.ku.edu/Publications/Bulletins/260/Bulletin_260_Dakota.pdf	-	-	Approximated from Fig. 3 of Whittlemore et al. (2014).
Picacho Basin	-	Pool, D. R., Carruth, R., Meehan, W. D. (2001). Hydrogeology of Picacho Basin, south-central Arizona. Water Resources Investigations Report 2000-4277, 72 pp. Accessed April 15, 2021 from https://pubs.usgs.gov/wri/2000/4277/report.pdf	-	-	Approximated from Fig. 2 of Pool et al. (2001).
Lake Mohave Basin	-	Tillman, F. D., Garner, B. D., Truini, M. (2013). Preliminary groundwater flow model of the basin fill aquifers in Detrital, Hualapai, and Sacramento Valleys, Mohave County, northwestern Arizona. U.S. Geological Survey Scientific	-	-	Approximated from Fig. 2 by Tillman et al. (2013).

Aquifer System	Broader System	References			Delineation steps and sources
		Investigations Report 2013–5122, 52 pp. http://pubs.usgs.gov/sir/2013/5122/			
Detrital Valley Basin	-	Tillman, F.D., Garner, B.D., Truini, M. (2013). Preliminary groundwater flow model of the basin-fill aquifers in Detrital, Hualapai, and Sacramento Valleys, Mohave County, northwestern Arizona. U.S. Geological Survey Scientific Investigations Report 2013–5122, 52 pp. http://pubs.usgs.gov/sir/2013/5122/	-	-	Approximated from Fig. 2 by Tillman et al. (2013).
Sacramento Valley Basin	-	Tillman, F.D., Garner, B.D., Truini, M. (2013). Preliminary groundwater flow model of the basin-fill aquifers in Detrital, Hualapai, and Sacramento Valleys, Mohave County, northwestern Arizona. U.S. Geological Survey Scientific Investigations Report 2013–5122, 52 pp. http://pubs.usgs.gov/sir/2013/5122/	-	-	Approximated from Fig. 2 by Tillman et al. (2013).
Hualapai Valley Basin	-	Tillman, F.D., Garner, B.D., Truini, M. (2013). Preliminary groundwater flow model of the basin-fill aquifers in Detrital, Hualapai, and Sacramento Valleys, Mohave County, northwestern Arizona. U.S. Geological Survey Scientific Investigations Report 2013–5122, 52 pp. http://pubs.usgs.gov/sir/2013/5122/	-	-	Approximated from Fig. 2 by Tillman et al. (2013).
Big Sandy Valley	-	Morrison, R.B. (1940). Ground-water resources of the Big Sandy Valley, Mohave County, Arizona. U.S. Geological Survey Report, 8 pp. Accessed April 15, 2021 from https://azmemory.azlibrary.gov/digital/collection/statepubs/id/6455/	-	-	Approximated from “Map of Big Sandy Valley” on page 8 of Morrison (1940).
McMullen Valley	-	Kam, W. (1964). Geology and Ground-Water Resources of McMullen Valley Maricopa, Yavapai, and Yuma Counties, Arizona. U.S. Geological Survey Water-Supply Paper 1665, 70 pp. Accessed April 15, 2021 from https://pubs.usgs.gov/wsp/1665/report.pdf	-	-	Approximated from Fig. 2 of Kam (1964).
Harquahala Basin	-	Tillman, F.D., Cordova, J.T., Leake, S.A., Thomas, B.E., Callegary, J.B. (2011). Water availability and use pilot: methods development for a regional assessment of groundwater availability, southwest alluvial basins, Arizona. U.S. Geological Survey Scientific Investigations Report	-	-	Approximated from Fig. 2 of Tillman et al. (2011); boundaries also informed by the density of wells recorded in Arizona’s well completion dataset.

Aquifer System	Broader System	References	Delineation steps and sources
Ranegras Plain	-	2011-5071, 132 pp. Accessed April 15, 2021 from https://pubs.usgs.gov/sir/2011/5071/sir2011-5071_text.pdf Tillman, F.D., Cordova, J.T., Leake, S.A., Thomas, B.E., Callegary, J.B. (2011). Water availability and use pilot: methods development for a regional assessment of groundwater availability, southwest alluvial basins, Arizona. U.S. Geological Survey Scientific Investigations Report 2011-5071, 132 pp. Accessed April 15, 2021 from https://pubs.usgs.gov/sir/2011/5071/sir2011-5071_text.pdf	Approximated from Fig. 2 of Tillman et al. (2011); boundaries also informed by the density of wells recorded in Arizona's well completion dataset.
Butler Valley	-	Tillman, F.D., Cordova, J.T., Leake, S.A., Thomas, B.E., Callegary, J.B. (2011). Water availability and use pilot: methods development for a regional assessment of groundwater availability, southwest alluvial basins, Arizona. U.S. Geological Survey Scientific Investigations Report 2011-5071, 132 pp. Accessed April 15, 2021 from https://pubs.usgs.gov/sir/2011/5071/sir2011-5071_text.pdf	Approximated from Fig. 2 of Tillman et al. (2011); boundaries also informed by the density of wells recorded in Arizona's well completion dataset.
Lower Gila Basin	-	Tillman, F.D., Cordova, J.T., Leake, S.A., Thomas, B.E., Callegary, J.B. (2011). Water availability and use pilot: methods development for a regional assessment of groundwater availability, southwest alluvial basins, Arizona. U.S. Geological Survey Scientific Investigations Report 2011-5071, 132 pp. Accessed April 15, 2021 from https://pubs.usgs.gov/sir/2011/5071/sir2011-5071_text.pdf	Approximated from Fig. 2 of Tillman et al. (2011); boundaries also informed by the density of wells recorded in Arizona's well completion dataset.
Gila Bend Basin	-	Tillman, F.D., Cordova, J.T., Leake, S.A., Thomas, B.E., Callegary, J.B. (2011). Water availability and use pilot: methods development for a regional assessment of groundwater availability, southwest alluvial basins, Arizona. U.S. Geological Survey Scientific Investigations Report 2011-5071, 132 pp. Accessed April 15, 2021 from https://pubs.usgs.gov/sir/2011/5071/sir2011-5071_text.pdf	Approximated from Fig. 2 of Tillman et al. (2011); boundaries also informed by the density of wells recorded in Arizona's well completion dataset.

Aquifer System	Broader System	References			Delineation steps and sources
Western C Aquifer	The C Aquifer	Brown, C.R., Macy, J.P. (2012). Groundwater, Surface Water, and Water Chemistry Data from the C Aquifer Monitoring Program, Northeastern Arizona, 2005–2011. U.S. Geological Survey Open-File Report 2012–1196, 46 pp. Accessed April 15, 2021 from https://pubs.usgs.gov/of/2012/1196/of2012-1196.pdf	-	-	Approximated from Fig. 1 of Brown and Macy (2012). East and west portions divided near Joseph City.
Eastern C Aquifer	The C Aquifer	Brown, C.R., Macy, J.P. (2012). Groundwater, Surface Water, and Water Chemistry Data from the C Aquifer Monitoring Program, Northeastern Arizona, 2005–2011. U.S. Geological Survey Open-File Report 2012–1196, 46 pp. Accessed April 15, 2021 from https://pubs.usgs.gov/of/2012/1196/of2012-1196.pdf	-	-	Approximated from Fig. 1 of Brown and Macy (2012). East and west portions divided near Joseph City.
Western Champlain Valley Lowlands	-	Nystrom, E.A. (2006). Ground water quality in the Lake Champlain Basin, New York, 2004. U.S. Geological Survey Open-File Report 2006–1088, 22 pp. Accessed April 15, 2021 from https://pubs.usgs.gov/of/2006/1088/pdf/Nystrom.OFR2006-1088.pdf	Scott, T., M., Nystrom, E.A., Reddy, J.E. (2016). Groundwater quality in the Lake Champlain and Susquehanna River basins, New York, 2014. U.S. Geological Survey Open-File Report 2016–1153, 33 pp. Accessed April 15, 2021 from https://pubs.usgs.gov/of/2016/1153/of20161153.pdf	-	Approximated from Fig. 2 of Nystrom (2006) and Fig. 1 by Scott et al. (2016).
Eastern Champlain Valley Lowlands	-	Nystrom, E.A. (2006). Ground water quality in the Lake Champlain Basin, New York, 2004. U.S. Geological Survey Open-File Report 2006–1088, 22 pp. Accessed April 15, 2021 from https://pubs.usgs.gov/of/2006/1088/pdf/Nystrom.OFR2006-1088.pdf	Scott, T., M., Nystrom, E.A., Reddy, J.E. (2016). Groundwater quality in the Lake Champlain and Susquehanna River basins, New York, 2014. U.S. Geological Survey Open-File Report 2016–1153, 33 pp. Accessed April 15, 2021 from https://pubs.usgs.gov/of/2016/1153/of20161153.pdf	-	Approximated from Fig. 2 of Nystrom (2006) and Fig. 1 by Scott et al. (2016).
Coastal Maine Crystalline Aquifers	-	Harte, P.T., Robinson, G.R., Jr., Ayotte, J.D., Flanagan, S.F. (2008). Framework for evaluating water quality of the New England crystalline rock aquifers. U.S. Geological Survey Open-File Report 2008–1282, 47 pp. Accessed April 15, 2021 from https://pubs.usgs.gov/of/2008/1282/pdf/ofr2008-1282.pdf	-	-	Approximated from Fig. 4 and "Coastal Maine" geologic province by Harte et al. (2008).
New Hampshire Seacoast	-	Mack, T.J. (2008). Assessment of Ground-Water Resources in the Seacoast Region of New Hampshire.	-	-	Approximated from Fig. 7 of Mack (2008)

Aquifer System	Broader System	References		Delineation steps and sources	
		Scientific Investigations Report 2008-5222, 66 pp. Accessed April 15, 2024 from https://pubs.usgs.gov/sir/2008/5222/pdf/sir2008-5222.pdf			
Paskapoo-Scollard	-	Grasby, S. E., Chen, Z., Hamblin, A. P., Wozniak, P. R., Sweet, A. R. (2008). Regional characterization of the Paskapoo bedrock aquifer system, southern Alberta. Canadian Journal of Earth Sciences, 45, 1501-1516.	Parks, K., Andriashek, L. (2009). Preliminary investigation of potential, natural hydraulic pathways between the Scollard and Paskapoo formations in Alberta: implications for coalbed methane production. ERCB/AGS Open File Report 2009-16, 66 pp. Accessed February 2, 2024 from https://static.ags.aer.ca/files/document/nl/OFR/OFR_2009_16.pdf	-	Approximated from Fig. 2 of Grasby et al. 2008, and Fig. 4 of Parks and Andriashek (2009)
American and Auburn Basins	Sierra Nevada Foothills	Shelton, J.L., Fram, M.S., Munday, C.M., Beltz, K. (2010). Groundwater quality data for the Sierra Nevada study unit, 2008. Results from the California GAMA program. U.S. Geological Survey Data Series 534, 106 pp. Accessed April 15, 2024 from https://pubs.usgs.gov/ds/534/ds_534.pdf	-	-	Approximated on the basis of the density of completed wells (from the California Department of Water Resources); general hydrogeology of region summarized by Shelton et al. (2010).
Butte Creek Basin	Sierra Nevada Foothills	Shelton, J.L., Fram, M.S., Munday, C.M., Beltz, K. (2010). Groundwater quality data for the Sierra Nevada study unit, 2008. Results from the California GAMA program. U.S. Geological Survey Data Series 534, 106 pp. Accessed April 15, 2024 from https://pubs.usgs.gov/ds/534/ds_534.pdf	-	-	Approximated on the basis of the density of completed wells (from the California Department of Water Resources); general hydrogeology of region summarized by Shelton et al. (2010).
Feather River Basin	Sierra Nevada Foothills	Shelton, J.L., Fram, M.S., Munday, C.M., Beltz, K. (2010). Groundwater quality data for the Sierra Nevada study unit, 2008. Results from the California GAMA program. U.S. Geological Survey Data Series 534, 106 pp. Accessed April 15, 2024 from https://pubs.usgs.gov/ds/534/ds_534.pdf	-	-	Approximated on the basis of the density of completed wells (from the California Department of Water Resources); general hydrogeology of region summarized by Shelton et al. (2010).
Yuba Basin	Sierra Nevada Foothills	Shelton, J.L., Fram, M.S., Munday, C.M., Beltz, K. (2010). Groundwater quality data for the Sierra Nevada study unit, 2008. Results from the California GAMA program. U.S. Geological Survey Data Series 534, 106 pp. Accessed April 15, 2024 from https://pubs.usgs.gov/ds/534/ds_534.pdf	-	-	Approximated on the basis of the density of completed wells (from the California Department of Water Resources); general hydrogeology of region summarized by Shelton et al. (2010).

Aquifer System	Broader System	References			Delineation steps and sources
Upper-Bear Basin	Sierra Nevada Foothills	Shelton, J.L., Fram, M.S., Munday, C.M., Belitz, K. (2010). Groundwater quality data for the Sierra Nevada study unit, 2008. Results from the California GAMA program. U.S. Geological Survey Data Series 534, 106 pp. Accessed April 15, 2021 from https://pubs.usgs.gov/ds/534/ds_534.pdf	-	-	Approximated on the basis of the density of completed wells (from the California Department of Water Resources); general hydrogeology of region summarized by Shelton et al. (2010).
Cosumnes Basin	Sierra Nevada Foothills	Shelton, J.L., Fram, M.S., Munday, C.M., Belitz, K. (2010). Groundwater quality data for the Sierra Nevada study unit, 2008. Results from the California GAMA program. U.S. Geological Survey Data Series 534, 106 pp. Accessed April 15, 2021 from https://pubs.usgs.gov/ds/534/ds_534.pdf	-	-	Approximated on the basis of the density of completed wells (from the California Department of Water Resources); general hydrogeology of region summarized by Shelton et al. (2010).
Mokelumne Basin	Sierra Nevada Foothills	Shelton, J.L., Fram, M.S., Munday, C.M., Belitz, K. (2010). Groundwater quality data for the Sierra Nevada study unit, 2008. Results from the California GAMA program. U.S. Geological Survey Data Series 534, 106 pp. Accessed April 15, 2021 from https://pubs.usgs.gov/ds/534/ds_534.pdf	-	-	Approximated on the basis of the density of completed wells (from the California Department of Water Resources); general hydrogeology of region summarized by Shelton et al. (2010).
Calaveras Basin	Sierra Nevada Foothills	Shelton, J.L., Fram, M.S., Munday, C.M., Belitz, K. (2010). Groundwater quality data for the Sierra Nevada study unit, 2008. Results from the California GAMA program. U.S. Geological Survey Data Series 534, 106 pp. Accessed April 15, 2021 from https://pubs.usgs.gov/ds/534/ds_534.pdf	-	-	Approximated on the basis of the density of completed wells (from the California Department of Water Resources); general hydrogeology of region summarized by Shelton et al. (2010).
Stanislaus and Rock Creek Basins	Sierra Nevada Foothills	Shelton, J.L., Fram, M.S., Munday, C.M., Belitz, K. (2010). Groundwater quality data for the Sierra Nevada study unit, 2008. Results from the California GAMA program. U.S. Geological Survey Data Series 534, 106 pp. Accessed April 15, 2021 from https://pubs.usgs.gov/ds/534/ds_534.pdf	-	-	Approximated on the basis of the density of completed wells (from the California Department of Water Resources); general hydrogeology of region summarized by Shelton et al. (2010).
Tuolumne Basin	Sierra Nevada Foothills	Shelton, J.L., Fram, M.S., Munday, C.M., Belitz, K. (2010). Groundwater quality data for the Sierra Nevada study unit, 2008. Results from the California GAMA program. U.S. Geological Survey Data Series 534, 106 pp. Accessed April 15, 2021 from https://pubs.usgs.gov/ds/534/ds_534.pdf	-	-	Approximated on the basis of the density of completed wells (from the California Department of Water Resources); general hydrogeology of region summarized by Shelton et al. (2010).

Aquifer System	Broader System	References			Delineation steps and sources
Merced Basin	Sierra Nevada Foothills	Shelton, J.L., Fram, M.S., Munday, C.M., Belitz, K. (2010). Groundwater quality data for the Sierra Nevada study unit, 2008. Results from the California GAMA program. U.S. Geological Survey Data Series 534, 106 pp. Accessed April 15, 2021 from https://pubs.usgs.gov/ds/534/ds_534.pdf	-	-	Approximated on the basis of the density of completed wells (from the California Department of Water Resources); general hydrogeology of region summarized by Shelton et al. (2010).
Fresno Basin	Sierra Nevada Foothills	Shelton, J.L., Fram, M.S., Munday, C.M., Belitz, K. (2010). Groundwater quality data for the Sierra Nevada study unit, 2008. Results from the California GAMA program. U.S. Geological Survey Data Series 534, 106 pp. Accessed April 15, 2021 from https://pubs.usgs.gov/ds/534/ds_534.pdf	-	-	Approximated on the basis of the density of completed wells (from the California Department of Water Resources); general hydrogeology of region summarized by Shelton et al. (2010).
Chowchilla Basin	Sierra Nevada Foothills	Shelton, J.L., Fram, M.S., Munday, C.M., Belitz, K. (2010). Groundwater quality data for the Sierra Nevada study unit, 2008. Results from the California GAMA program. U.S. Geological Survey Data Series 534, 106 pp. Accessed April 15, 2021 from https://pubs.usgs.gov/ds/534/ds_534.pdf	-	-	Approximated on the basis of the density of completed wells (from the California Department of Water Resources); general hydrogeology of region summarized by Shelton et al. (2010).
Upper San Joaquin	Sierra Nevada Foothills	Shelton, J.L., Fram, M.S., Munday, C.M., Belitz, K. (2010). Groundwater quality data for the Sierra Nevada study unit, 2008. Results from the California GAMA program. U.S. Geological Survey Data Series 534, 106 pp. Accessed April 15, 2021 from https://pubs.usgs.gov/ds/534/ds_534.pdf	-	-	Approximated on the basis of the density of completed wells (from the California Department of Water Resources); general hydrogeology of region summarized by Shelton et al. (2010).
Dry and King Basins	Sierra Nevada Foothills	Shelton, J.L., Fram, M.S., Munday, C.M., Belitz, K. (2010). Groundwater quality data for the Sierra Nevada study unit, 2008. Results from the California GAMA program. U.S. Geological Survey Data Series 534, 106 pp. Accessed April 15, 2021 from https://pubs.usgs.gov/ds/534/ds_534.pdf	-	-	Approximated on the basis of the density of completed wells (from the California Department of Water Resources); general hydrogeology of region summarized by Shelton et al. (2010).
Kaweah Basin	Sierra Nevada Foothills	Shelton, J.L., Fram, M.S., Munday, C.M., Belitz, K. (2010). Groundwater quality data for the Sierra Nevada study unit, 2008. Results from the California GAMA program. U.S. Geological Survey Data Series 534, 106 pp. Accessed April 15, 2021 from https://pubs.usgs.gov/ds/534/ds_534.pdf	-	-	Approximated on the basis of the density of completed wells (from the California Department of Water Resources); general hydrogeology of region summarized by Shelton et al. (2010).

Aquifer System	Broader System	References			Delineation steps and sources
Tule Basin	Sierra Nevada Foothills	Shelton, J.L., Fram, M.S., Munday, C.M., Belitz, K. (2010). Groundwater quality data for the Sierra Nevada study unit, 2008. Results from the California GAMA program. U.S. Geological Survey Data Series 534, 106 pp. Accessed April 15, 2024 from https://pubs.usgs.gov/ds/534/ds_534.pdf	-	-	Approximated on the basis of the density of completed wells (from the California Department of Water Resources); general hydrogeology of region summarized by Shelton et al. (2010).
Deer and White Basins	Sierra Nevada Foothills	Shelton, J.L., Fram, M.S., Munday, C.M., Belitz, K. (2010). Groundwater quality data for the Sierra Nevada study unit, 2008. Results from the California GAMA program. U.S. Geological Survey Data Series 534, 106 pp. Accessed April 15, 2024 from https://pubs.usgs.gov/ds/534/ds_534.pdf	-	-	Approximated on the basis of the density of completed wells (from the California Department of Water Resources); general hydrogeology of region summarized by Shelton et al. (2010).
Poso Basin	Sierra Nevada Foothills	Shelton, J.L., Fram, M.S., Munday, C.M., Belitz, K. (2010). Groundwater quality data for the Sierra Nevada study unit, 2008. Results from the California GAMA program. U.S. Geological Survey Data Series 534, 106 pp. Accessed April 15, 2024 from https://pubs.usgs.gov/ds/534/ds_534.pdf	-	-	Approximated on the basis of the density of completed wells (from the California Department of Water Resources); general hydrogeology of region summarized by Shelton et al. (2010).

Supplementary Note 5. Well depth versus modern groundwater plots for study aquifers

The following pages provide a set of plots showing how modern groundwater varies with well depth; each plot represents a single aquifer system (i.e., all groundwater tritium measurements collected from within the boundaries of a given aquifer system).

The title of the aquifer System/system is identified in the figure caption. Each yellow circle represents the calculated maximum fraction of the well water sample comprised of modern water (yellow circles correspond to the x-axis labels on the bottom of the plot).

The blue line presents the proportion of all groundwater samples collected from wells deeper than a given depth that contain minimal (<25%) modern water; we only present a blue line if at least five modern groundwater fraction data points are available for deeper wells (i.e., deeper than any given well).

The three diamonds represent the depths at which one of the following criteria are met:

- **Red diamond** – the depth below which >60% of well water samples contain less than 25% modern water
- **Purple Diamond** – the depth below which >70% of well water samples contain less than 25% modern water
- **Green Diamond** – the depth below which >80% of well water samples contain less than 25% modern water

The y-axis on each plot extends to a depth of 1,000 m below land surface; however, a small number of groundwater ³H samples were collected from wells with depths exceeding 1,000 m, and are thus not plotted in Supplementary Figs. 8-98. There are only n=5 samples in our dataset collected from wells with depths exceeding 1,000 m (representing 0.05% of all of the n=9,333 groundwater tritium samples plotted in Supplementary Figs. 8-98). These n=5 samples are located in the Yakima Basin (n=1 sample from a well with a depth exceeding 1,000 m), the Western Carrizo-Wilcox (n=1 sample), the Lower Coastal Plain (n=1 sample) and the Ozark Plateaus Aquifer System (n=2 samples).

Formatted: Line spacing: At least 1.15 pt

Supplementary Fig. 8. Sacramento Valley modern well water prevalence with depth. For details on symbology see the paragraph at the beginning of Supplementary Note 5.

Supplementary Fig. 9. San Joaquin Basin modern well water prevalence with depth. For details on symbology see the paragraph at the beginning of Supplementary Note 5.

Supplementary Fig. 10. Tulare Basin modern well water prevalence with depth. For details on symbology see the paragraph at the beginning of Supplementary Note 5.

Supplementary Fig. 11. Central Carrizo Wilcox modern well water prevalence with depth.
For details on symbology see the paragraph at the beginning of Supplementary Note 5.

Supplementary Fig. 12. Eastern Carrizo-Wilcox modern well water prevalence with depth.
For details on symbology see the paragraph at the beginning of Supplementary Note 5.

Supplementary Fig. 13. Western Carrizo-Wilcox modern well water prevalence with depth.
For details on symbology see the paragraph at the beginning of Supplementary Note 5.

Supplementary Fig. 14. Eagle Valley modern well water prevalence with depth. For details on symbology see the paragraph at the beginning of Supplementary Note 5.

Supplementary Fig. 15. Central Wabash and Bloomington Ridged Plain modern well water prevalence with depth. For details on symbology see the paragraph at the beginning of Supplementary Note 5.

Supplementary Fig. 16. Blue Mountains and Clearwater Embayment modern well water prevalence with depth. For details on symbology see the paragraph at the beginning of Supplementary Note 5.

Supplementary Fig. 17. Palouse Slope modern well water prevalence with depth. For details on symbology see the paragraph at the beginning of Supplementary Note 5.

Supplementary Fig. 18. Umatilla Basin and Horse Heaven Hills modern well water prevalence with depth. For details on symbology see the paragraph at the beginning of Supplementary Note 5.

Supplementary Fig. 19. Walla Walla Basin modern well water prevalence with depth. For details on symbology see the paragraph at the beginning of Supplementary Note 5.

Supplementary Fig. 20. Yakima Basin modern well water prevalence with depth. For details on symbology see the paragraph at the beginning of Supplementary Note 5.

Supplementary Fig. 21. Stockton Plateau modern well water prevalence with depth. For details on symbology see the paragraph at the beginning of Supplementary Note 5.

Supplementary Fig. 22. Trinity Aquifer System modern well water prevalence with depth.
For details on symbology see the paragraph at the beginning of Supplementary Note 5.

Supplementary Fig. 23. Bacon Terrace modern well water prevalence with depth. For details on symbology see the paragraph at the beginning of Supplementary Note 5.

Supplementary Fig. 24. Dougherty Plain and Marianna Lowlands modern well water prevalence with depth. For details on symbology see the paragraph at the beginning of Supplementary Note 5.

Supplementary Fig. 25. Eastern Flatwoods Southshores modern well water prevalence with depth. For details on symbology see the paragraph at the beginning of Supplementary Note 5.

Supplementary Fig. 26. Intermediate Aquifer modern well water prevalence with depth.
For details on symbology see the paragraph at the beginning of Supplementary Note 5.

Supplementary Fig. 27. Lower Coastal Plain modern well water prevalence with depth.
For details on symbology see the paragraph at the beginning of Supplementary Note 5.

Supplementary-Fig.-28. Ocala Uplift modern well water prevalence with depth. For details on symbology see the paragraph at the beginning of Supplementary Note 5.

Supplementary Fig. 29. Sea Island modern well water prevalence with depth. For details on symbology see the paragraph at the beginning of Supplementary Note 5.

Supplementary Fig. 30. Tifton Upland modern well water prevalence with depth. For details on symbology see the paragraph at the beginning of Supplementary Note 5.

Supplementary Fig. 31. Vidalia Upland modern well water prevalence with depth. For details on symbology see the paragraph at the beginning of Supplementary Note 5.

Supplementary Fig. 32. Catahoula Area modern well water prevalence with depth. For details on symbology see the paragraph at the beginning of Supplementary Note 5.

Supplementary Fig. 33. Houston-Galveston Area modern well water prevalence with depth. For details on symbology see the paragraph at the beginning of Supplementary Note 5.

Supplementary Fig. 34. Lafayette Area modern well water prevalence with depth. For details on symbology see the paragraph at the beginning of Supplementary Note 5.

Supplementary Fig. 35. Southern Hills modern well water prevalence with depth. For details on symbology see the paragraph at the beginning of Supplementary Note 5.

Supplementary Fig. 36. Central High Plains modern well water prevalence with depth. For details on symbology see the paragraph at the beginning of Supplementary Note 5.

Supplementary Fig. 37. Northern High Plains modern well water prevalence with depth.
For details on symbology see the paragraph at the beginning of Supplementary Note 5.

Supplementary Fig. 38. Southern High Plains modern well water prevalence with depth.
For details on symbology see the paragraph at the beginning of Supplementary Note 5.

Supplementary Fig. 39. Albuquerque Basin modern well water prevalence with depth. For details on symbology see the paragraph at the beginning of Supplementary Note 5.

Supplementary Fig. 40. Espanola Basin modern well water prevalence with depth. For details on symbology see the paragraph at the beginning of Supplementary Note 5.

Supplementary Fig. 41. San Luis Valley modern well water prevalence with depth. For details on symbology see the paragraph at the beginning of Supplementary Note 5.

Supplementary Fig. 42. Central Mississippi Embayment modern well water prevalence with depth. For details on symbology see the paragraph at the beginning of Supplementary Note 5.

Supplementary Fig. 43. Eastern Mississippi Embayment modern well-water prevalence with depth. For details on symbology see the paragraph at the beginning of Supplementary Note 5.

Supplementary Fig. 44. Western Mississippi Embayment modern well water prevalence with depth. For details on symbology see the paragraph at the beginning of Supplementary Note 5.

Supplementary Fig. 45. Dolmarva Peninsula modern well water prevalence with depth.
 For details on symbology see the paragraph at the beginning of Supplementary Note 5.

Supplementary Fig. 46. Maryland Western Shores modern well water prevalence with depth. For details on symbology see the paragraph at the beginning of Supplementary Note 5.

Supplementary Fig. 47. New Jersey Coastal Plain modern well water prevalence with depth. For details on symbology see the paragraph at the beginning of Supplementary Note 5.

Supplementary Fig. 48. North Carolina and Virginia Coastal Plain modern well water prevalence with depth. For details on symbology see the paragraph at the beginning of Supplementary Note 5.

Supplementary Fig. 49. Powder River Basin modern well water prevalence with depth. For details on symbology see the paragraph at the beginning of Supplementary Note 5.

Supplementary Fig. 50. Williston Basin modern well water prevalence with depth. For details on symbology see the paragraph at the beginning of Supplementary Note 5.

Supplementary Fig. 51. Eastern Cambrian-Ordovician Aquifers modern well water prevalence with depth. For details on symbology see the paragraph at the beginning of Supplementary Note 5.

Supplementary Fig. 52. Eastern Silurian-Devonian Aquifers modern well water prevalence with depth. For details on symbology see the paragraph at the beginning of Supplementary Note 5.

Supplementary Fig. 53. Mississippian-Silurian-Devonian Carbonates modern well water prevalence with depth. For details on symbology see the paragraph at the beginning of Supplementary Note 5.

Supplementary Fig. 54. Northeast Missouri Carbonates modern well water prevalence with depth. For details on symbology see the paragraph at the beginning of Supplementary Note 5.

Supplementary Fig. 55. Northern Cambrian-Ordovician Aquifers modern well water prevalence with depth. For details on symbology see the paragraph at the beginning of Supplementary Note 5.

Supplementary Fig. 56. Upper Carbonate Aquifer modern well water prevalence with depth. For details on symbology see the paragraph at the beginning of Supplementary Note 5.

Supplementary Fig. 57. Western Cambrian-Ordovician Aquifers modern well water prevalence with depth. For details on symbology see the paragraph at the beginning of Supplementary Note 5.

Supplementary Fig. 58. Mesilla Valley modern well water prevalence with depth. For details on symbology see the paragraph at the beginning of Supplementary Note 5.

Supplementary Fig. 59. West Salt River Basin modern well water prevalence with depth.
For details on symbology see the paragraph at the beginning of Supplementary Note 5.

Supplementary Fig. 60. Lower Santa Ynez Valley modern well water prevalence with depth. For details on symbology see the paragraph at the beginning of Supplementary Note 5.

Supplementary Fig. 61. Valle de Juarez and Hueco Bolson modern well water prevalence with depth. For details on symbology see the paragraph at the beginning of Supplementary Note 5.

Supplementary Fig. 62. Boise Valley and Homedale-Murphy Area modern well water prevalence with depth. For details on symbology see the paragraph at the beginning of Supplementary Note 5.

Supplementary Fig. 63. Mountain Home Plateau modern well water prevalence with depth. For details on symbology see the paragraph at the beginning of Supplementary Note 5.

Supplementary Fig. 64. Antelope Valley modern well water prevalence with depth. For details on symbology see the paragraph at the beginning of Supplementary Note 5.

Supplementary Fig. 65. Big Bear Valley modern well water prevalence with depth. For details on symbology see the paragraph at the beginning of Supplementary Note 5.

Supplementary Fig. 66. Big Chino Valley modern well water prevalence with depth. For details on symbology see the paragraph at the beginning of Supplementary Note 5.

Supplementary Fig. 67. Bighorn Basin modern well water prevalence with depth. For details on symbology see the paragraph at the beginning of Supplementary Note 5.

Supplementary Fig. 68. Black Hills Uplift modern well water prevalence with depth. For details on symbology see the paragraph at the beginning of Supplementary Note 5.

Supplementary Fig.-69. Black Warrior River Aquifer System (Eutaw and McShan Formations and Tuscaloosa Group) modern well water prevalence with depth. For details on symbology see the paragraph at the beginning of Supplementary Note 5.

Supplementary Fig. 70. Castle Hayne Aquifer modern well water prevalence with depth.
For details on symbology see the paragraph at the beginning of Supplementary Note 5.

Supplementary Fig. 71. Coachella Valley modern well water prevalence with depth. For details on symbology see the paragraph at the beginning of Supplementary Note 5.

Supplementary Fig. 72. Cuyama Valley modern well water prevalence with depth. For details on symbology see the paragraph at the beginning of Supplementary Note 5.

Supplementary Fig. 73. Denver Basin modern well water prevalence with depth. For details on symbology see the paragraph at the beginning of Supplementary Note 5.

Supplementary Fig. 74. Eastern Dakota Aquifer modern well water prevalence with depth. For details on symbology see the paragraph at the beginning of Supplementary Note 5.

Supplementary Fig. 75. Eureka and Eel River and Mad River Plains modern well water prevalence with depth. For details on symbology see the paragraph at the beginning of Supplementary Note 5.

Supplementary Fig. 76. Garber-Wellington Aquifer modern well water prevalence with depth. For details on symbology see the paragraph at the beginning of Supplementary Note 5.

Supplementary Fig. 77. Honey Lake Valley modern well water prevalence with depth. For details on symbology see the paragraph at the beginning of Supplementary Note 5.

Supplementary Fig. 78. Judith Basin modern well water prevalence with depth. For details on symbology see the paragraph at the beginning of Supplementary Note 5.

Supplementary Fig. 79. Long Island modern well water prevalence with depth. For details on symbology see the paragraph at the beginning of Supplementary Note 5.

Supplementary Fig. 80. Los Angeles Basin modern well water prevalence with depth. For details on symbology see the paragraph at the beginning of Supplementary Note 5.

Supplementary Fig. 81. Michigan Basin modern well water prevalence with depth. For details on symbology see the paragraph at the beginning of Supplementary Note 5.

Supplementary Fig. 82. Mojave Basin modern well water prevalence with depth. For details on symbology see the paragraph at the beginning of Supplementary Note 5.

Supplementary Fig. 83. Northern Green River Basin modern well-water prevalence with depth. For details on symbology see the paragraph at the beginning of Supplementary Note 5.

Supplementary Fig. 84. Ozark Plateaus Aquifer System modern well water prevalence with depth. For details on symbology see the paragraph at the beginning of Supplementary Note 5.

Supplementary Fig. 85. Pearl and Chattahoochee Aquifer System modern well water prevalence with depth. For details on symbology see the paragraph at the beginning of Supplementary Note 5.

Supplementary Fig. 86. Peebles and Black Creek and Cape Fear Aquifers modern well water prevalence with depth. For details on symbology see the paragraph at the beginning of Supplementary Note 5.

Supplementary Fig. 87. Salinas Valley modern well water prevalence with depth. For details on symbology see the paragraph at the beginning of Supplementary Note 5.

Supplementary Fig. 88. Salt Lake Valley modern well water prevalence with depth. For details on symbology see the paragraph at the beginning of Supplementary Note 5.

Supplementary Fig. 89. San Antonio Creek Valley modern well water prevalence with depth. For details on symbology see the paragraph at the beginning of Supplementary Note 5.

Supplementary Fig. 90. San Pedro Basin modern well water prevalence with depth. For details on symbology see the paragraph at the beginning of Supplementary Note 5.

Supplementary Fig. 91. Santa Clara-Calleguas Basin modern well water prevalence with depth. For details on symbology see the paragraph at the beginning of Supplementary Note 5.

Supplementary Fig. 92. Santa Rosa Valley modern well water prevalence with depth. For details on symbology see the paragraph at the beginning of Supplementary Note 5.

Supplementary Fig. 93. South Park Basin modern well water prevalence with depth. For details on symbology see the paragraph at the beginning of Supplementary Note 5.

Supplementary Fig. 94. Southern San Juan Basin modern well water prevalence with depth. For details on symbology see the paragraph at the beginning of Supplementary Note 5.

Supplementary Fig.-95. Spanish Springs Valley modern well water prevalence with depth.
For details on symbology see the paragraph at the beginning of Supplementary Note 5.

Supplementary Fig. 96. Tijuana-San Diego modern well water prevalence with depth. For details on symbology see the paragraph at the beginning of Supplementary Note 5.

Supplementary Fig.-97. Upper Santa Ana Basin modern well water prevalence with depth.
For details on symbology see the paragraph at the beginning of Supplementary Note 5.

Supplementary Fig. 98. Utah Lake Valley modern well water prevalence with depth. For details on symbology see the paragraph at the beginning of Supplementary Note 5.

~~Supplementary Note 6. Sensitivity analyses of statistical relationship between groundwater withdrawals and the depth that modern water reaches after visual inspection of scatterplots of depth versus modern groundwater~~

In most, but not all, of our aquifer systems, our method that identifies the depths below which over 60%, 70% or 80% of deeper wells contain minimal modern groundwater (<25%) appears to adequately capture the depth below which modern groundwaters become scarce (see diamonds in Supplementary Figs. 8-981-91). However, by plotting all n=91 study aquifer systems, we identified some cases where our method does not adequately capture the depth below which modern groundwaters become scarce for at least two broad reasons: (a) Well water tritium data are limited or absent for substantial well depth intervals, or (b) modern groundwater fractions decline to a certain depth, but increase at deeper depths.

These n=17 aquifer systems (where our method was deemed imperfect following visual inspection) are listed in Supplementary Table 9, and our resulting test for the robustness of findings presented in the main text (i.e., the main text results which included the results for the aquifer systems in Supplementary Table 9) when these aquifer systems are removed are detailed in Supplementary Table 10 and 11. We excluded these aquifer systems from further analyses and from the plots and correlations presented within the main text.

Supplementary Table 91. Aquifers where the depth below which over 60%, 70% or 80% of deeper wells contain minimal modern groundwater (<25%) appears an imperfect estimate of the depth below which modern groundwaters become scarce

Example	Aquifer Systems System	Supplementary Fig.
(a) Well water tritium data are limited or absent for substantial well depth intervals (i.e., y-axis values in Supplementary Figs. 8-981-91)	Western Mississippi Embayment	4437
	Valle de Juarez and Hueco Bolson	6154
	Blue Mountains and Clearwater Embayment	169
	Espanola Basin	4033
	Judith Basin	7871
	Peedee and Black Creek and Cape Fear Aquifers	8679
	San Antonio Creek Valley	8982
	Southern San Juan Basin	9487
	Bacon Terrace	2316
	Central Carrizo-Wilcox	144
	Spanish Springs Valley	9588
Big Chino Valley	6659	
(b) Modern groundwater fractions decline to a certain depth, but increase at deeper depths	West Salt River Basin	5952
	Eastern Cambrian-Ordovician Aquifers	5144
	Intermediate Aquifer	2619
	Western Carrizo-Wilcox	136
	Walla Walla Basin	1912

Formatted Table

Supplementary Fig. 1. Sacramento Valley modern well water prevalence with depth. For details on symbology see the paragraph at the beginning of Supplementary Note 1.

Formatted: Font: 11 pt

Supplementary Fig. 2. San Joaquin Basin modern well water prevalence with depth. For details on symbology see the paragraph at the beginning of Supplementary Note 1.

Formatted: Font: 11 pt, Font color: Text 1

Supplementary Fig. 3. Tulare Basin modern well water prevalence with depth. For details on symbology see the paragraph at the beginning of Supplementary Note 1.

Supplementary Fig. 4. Central Carrizo-Wilcox modern well water prevalence with depth.
For details on symbology see the paragraph at the beginning of Supplementary Note 1.

Supplementary Fig. 5. Eastern Carrizo-Wilcox modern well water prevalence with depth.
For details on symbology see the paragraph at the beginning of Supplementary Note 1.

Supplementary Fig. 6. Western Carrizo-Wilcox modern well water prevalence with depth.
For details on symbology see the paragraph at the beginning of Supplementary Note 1.

Supplementary Fig. 7. Eagle Valley modern well water prevalence with depth. For details on symbology see the paragraph at the beginning of Supplementary Note 1.

Supplementary Fig. 8. Central Wabash and Bloomington Ridged Plain modern well water prevalence with depth. For details on symbology see the paragraph at the beginning of Supplementary Note 1.

Supplementary Fig. 9. Blue Mountains and Clearwater Embayment modern well water prevalence with depth. For details on symbology see the paragraph at the beginning of Supplementary Note 1.

Supplementary Fig. 10. Palouse Slope modern well water prevalence with depth. For details on symbology see the paragraph at the beginning of Supplementary Note 1.

Supplementary Fig. 11. Umatilla Basin and Horse Heaven Hills modern well water prevalence with depth. For details on symbology see the paragraph at the beginning of Supplementary Note 1.

Supplementary Fig. 12. Walla Walla Basin modern well water prevalence with depth. For details on symbology see the paragraph at the beginning of Supplementary Note 1.

Supplementary Fig. 13. Yakima Basin modern well water prevalence with depth. For details on symbology see the paragraph at the beginning of Supplementary Note 1.

Supplementary Fig. 14. Stockton Plateau modern well water prevalence with depth. For details on symbology see the paragraph at the beginning of Supplementary Note 1.

Supplementary Fig. 15. Trinity Aquifer System modern well water prevalence with depth.
For details on symbology see the paragraph at the beginning of Supplementary Note 1.

Supplementary Fig. 16. Bacon Terrace modern well water prevalence with depth. For details on symbology see the paragraph at the beginning of Supplementary Note 1.

Supplementary Fig. 17. Dougherty Plain and Marianna Lowlands modern well water prevalence with depth. For details on symbology see the paragraph at the beginning of Supplementary Note 1.

Supplementary Fig. 18. Eastern Flatwoods Southshores modern well water prevalence with depth. For details on symbology see the paragraph at the beginning of Supplementary Note 1.

Supplementary Fig. 19. Intermediate Aquifer modern well water prevalence with depth.
For details on symbology see the paragraph at the beginning of Supplementary Note 1.

Supplementary Fig. 20. Lower Coastal Plain modern well water prevalence with depth.
For details on symbology see the paragraph at the beginning of Supplementary Note 1.

Supplementary Fig. 21. Ocala Uplift modern well water prevalence with depth. For details on symbology see the paragraph at the beginning of Supplementary Note 1.

Supplementary Fig. 22. Sea Island modern well water prevalence with depth. For details on symbology see the paragraph at the beginning of Supplementary Note 1.

Supplementary Fig. 23. Tifton Upland modern well water prevalence with depth. For details on symbology see the paragraph at the beginning of Supplementary Note 1.

Supplementary Fig. 24. Vidalia Upland modern well water prevalence with depth. For details on symbology see the paragraph at the beginning of Supplementary Note 1.

Supplementary Fig. 25. Catahoula Area modern well water prevalence with depth. For details on symbology see the paragraph at the beginning of Supplementary Note 1.

Supplementary Fig. 26. Houston-Galveston Area modern well water prevalence with depth. For details on symbology see the paragraph at the beginning of Supplementary Note 1.

Supplementary Fig. 27. Lafayette Area modern well water prevalence with depth. For details on symbology see the paragraph at the beginning of Supplementary Note 1.

Supplementary Fig. 28. Southern Hills modern well water prevalence with depth. For details on symbology see the paragraph at the beginning of Supplementary Note 1.

Supplementary Fig. 29. Central High Plains modern well water prevalence with depth. For details on symbology see the paragraph at the beginning of Supplementary Note 1.

Supplementary Fig. 30. Northern High Plains modern well water prevalence with depth.
For details on symbology see the paragraph at the beginning of Supplementary Note 1.

Supplementary Fig. 31. Southern High Plains modern well water prevalence with depth.
For details on symbology see the paragraph at the beginning of Supplementary Note 1.

Supplementary Fig. 32. Albuquerque Basin modern well water prevalence with depth. For details on symbology see the paragraph at the beginning of Supplementary Note 1.

Supplementary Fig. 33. Espanola Basin modern well water prevalence with depth. For details on symbology see the paragraph at the beginning of Supplementary Note 1.

Supplementary Fig. 34. San Luis Valley modern well water prevalence with depth. For details on symbology see the paragraph at the beginning of Supplementary Note 1.

Supplementary Fig. 35. Central Mississippi Embayment modern well water prevalence with depth. For details on symbology see the paragraph at the beginning of Supplementary Note 1.

Supplementary Fig. 36. Eastern Mississippi Embayment modern well water prevalence with depth. For details on symbology see the paragraph at the beginning of Supplementary Note 1.

Supplementary Fig. 37. Western Mississippi Embayment modern well water prevalence with depth. For details on symbology see the paragraph at the beginning of Supplementary Note 1.

Supplementary Fig. 38. Delmarva Peninsula modern well water prevalence with depth.
For details on symbology see the paragraph at the beginning of Supplementary Note 1.

Supplementary Fig. 39. Maryland Western Shores modern well water prevalence with depth. For details on symbology see the paragraph at the beginning of Supplementary Note 1.

Supplementary Fig. 40. New Jersey Coastal Plain modern well water prevalence with depth. For details on symbology see the paragraph at the beginning of Supplementary Note 1.

Supplementary Fig. 41. North Carolina and Virginia Coastal Plain modern well water prevalence with depth. For details on symbology see the paragraph at the beginning of Supplementary Note 1.

Supplementary Fig. 42. Powder River Basin modern well water prevalence with depth. For details on symbology see the paragraph at the beginning of Supplementary Note 1.

Supplementary Fig. 43. Williston Basin modern well water prevalence with depth. For details on symbology see the paragraph at the beginning of Supplementary Note 1.

Supplementary Fig. 44. Eastern Cambrian-Ordovician Aquifers modern well water prevalence with depth. For details on symbology see the paragraph at the beginning of Supplementary Note 1.

Supplementary Fig. 45. Eastern Silurian-Devonian Aquifers modern well water prevalence with depth. For details on symbology see the paragraph at the beginning of Supplementary Note 1.

Supplementary Fig. 46. Mississippian-Silurian-Devonian Carbonates modern well water prevalence with depth. For details on symbology see the paragraph at the beginning of Supplementary Note 1.

Supplementary Fig. 47. Northeast Missouri Carbonates modern well water prevalence with depth. For details on symbology see the paragraph at the beginning of Supplementary Note 1.

Supplementary Fig. 48. Northern Cambrian-Ordovician Aquifers modern well water prevalence with depth. For details on symbology see the paragraph at the beginning of Supplementary Note 1.

Supplementary Fig. 49. Upper Carbonate Aquifer modern well water prevalence with depth. For details on symbology see the paragraph at the beginning of Supplementary Note 1.

Supplementary Fig. 50. Western Cambrian-Ordovician Aquifers modern well water prevalence with depth. For details on symbology see the paragraph at the beginning of Supplementary Note 1.

Supplementary Fig. 51. Mesilla Valley modern well water prevalence with depth. For details on symbology see the paragraph at the beginning of Supplementary Note 1.

Supplementary Fig. 52. West Salt River Basin modern well water prevalence with depth.
For details on symbology see the paragraph at the beginning of Supplementary Note 1.

Supplementary Fig. 53. Lower Santa Ynez Valley modern well water prevalence with depth. For details on symbology see the paragraph at the beginning of Supplementary Note 1.

Supplementary Fig. 54. Valle de Juarez and Hueco Bolson modern well water prevalence with depth. For details on symbology see the paragraph at the beginning of Supplementary Note 1.

Supplementary Fig. 55. Boise Valley and Homedale Murphy Area modern well water prevalence with depth. For details on symbology see the paragraph at the beginning of Supplementary Note 1.

Supplementary Fig. 56. Mountain Home Plateau modern well water prevalence with depth. For details on symbology see the paragraph at the beginning of Supplementary Note 1.

Supplementary Fig. 57. Antelope Valley modern well water prevalence with depth. For details on symbology see the paragraph at the beginning of Supplementary Note 1.

Supplementary Fig. 58. Big Bear Valley modern well water prevalence with depth. For details on symbology see the paragraph at the beginning of Supplementary Note 1.

Supplementary Fig. 59. Big Chino Valley modern well water prevalence with depth. For details on symbology see the paragraph at the beginning of Supplementary Note 1.

Supplementary Fig. 60. Bighorn Basin modern well water prevalence with depth. For details on symbology see the paragraph at the beginning of Supplementary Note 1.

Supplementary Fig. 61. Black Hills Uplift modern well water prevalence with depth. For details on symbology see the paragraph at the beginning of Supplementary Note 1.

Supplementary Fig. 62. Black Warrior River Aquifer System (Eutaw and McShan Formations and Tuscaloosa Group) modern well water prevalence with depth. For details on symbology see the paragraph at the beginning of Supplementary Note 1.

Supplementary Fig. 63. Castle Hayne Aquifer modern well water prevalence with depth.
For details on symbology see the paragraph at the beginning of Supplementary Note 1.

Supplementary Fig. 64. Coachella Valley modern well water prevalence with depth. For details on symbology see the paragraph at the beginning of Supplementary Note 1.

Supplementary Fig. 65. Cuyama Valley modern well water prevalence with depth. For details on symbology see the paragraph at the beginning of Supplementary Note 1.

Supplementary Fig. 66. Denver Basin modern well water prevalence with depth. For details on symbology see the paragraph at the beginning of Supplementary Note 1.

Supplementary Fig. 67. Eastern Dakota Aquifer modern well water prevalence with depth.
For details on symbology see the paragraph at the beginning of Supplementary Note 1.

Supplementary Fig. 68. Eureka and Eel River and Mad River Plains modern well water prevalence with depth. For details on symbology see the paragraph at the beginning of Supplementary Note 1.

Supplementary Fig. 69. Garber-Wellington Aquifer modern well water prevalence with depth. For details on symbology see the paragraph at the beginning of Supplementary Note 1.

Supplementary Fig. 70. Honey Lake Valley modern well water prevalence with depth. For details on symbology see the paragraph at the beginning of Supplementary Note 1.

Supplementary Fig. 71. Judith Basin modern well water prevalence with depth. For details on symbology see the paragraph at the beginning of Supplementary Note 1.

Supplementary Fig. 72. Long Island modern well water prevalence with depth. For details on symbology see the paragraph at the beginning of Supplementary Note 1.

Supplementary Fig. 73. Los Angeles Basin modern well water prevalence with depth. For details on symbology see the paragraph at the beginning of Supplementary Note 1.

Supplementary Fig. 74. Michigan Basin modern well water prevalence with depth. For details on symbology see the paragraph at the beginning of Supplementary Note 1.

Supplementary Fig. 75. Mojave Basin modern well water prevalence with depth. For details on symbology see the paragraph at the beginning of Supplementary Note 1.

Supplementary Fig. 76. Northern Green River Basin modern well water prevalence with depth. For details on symbology see the paragraph at the beginning of Supplementary Note 1.

Supplementary Fig. 77. Ozark Plateaus Aquifer System modern well water prevalence with depth. For details on symbology see the paragraph at the beginning of Supplementary Note 1.

Supplementary Fig. 78. Pearl and Chattahoochee Aquifer System modern well water prevalence with depth. For details on symbology see the paragraph at the beginning of Supplementary Note 1.

Supplementary Fig. 79. Peedee and Black Creek and Cape Fear Aquifers modern well water prevalence with depth. For details on symbology see the paragraph at the beginning of Supplementary Note 1.

Supplementary Fig. 80. Salinas Valley modern well water prevalence with depth. For details on symbology see the paragraph at the beginning of Supplementary Note 1.

Supplementary Fig. 81. Salt Lake Valley modern well water prevalence with depth. For details on symbology see the paragraph at the beginning of Supplementary Note 1.

Supplementary Fig. 82. San Antonio Creek Valley modern well water prevalence with depth. For details on symbology see the paragraph at the beginning of Supplementary Note 1.

Supplementary Fig. 83. San Pedro Basin modern well water prevalence with depth. For details on symbology see the paragraph at the beginning of Supplementary Note 1.

Supplementary Fig. 84. Santa Clara-Calleguas Basin modern well water prevalence with depth. For details on symbology see the paragraph at the beginning of Supplementary Note 1.

Supplementary Fig. 85. Santa Rosa Valley modern well water prevalence with depth. For details on symbology see the paragraph at the beginning of Supplementary Note 1.

Supplementary Fig. 86. South Park Basin modern well water prevalence with depth. For details on symbology see the paragraph at the beginning of Supplementary Note 1.

Supplementary Fig. 87. Southern San Juan Basin modern well water prevalence with depth. For details on symbology see the paragraph at the beginning of Supplementary Note 1.

Supplementary Fig. 88. Spanish Springs Valley modern well water prevalence with depth.
For details on symbology see the paragraph at the beginning of Supplementary Note 1.

Supplementary Fig. 89. Tijuana-San Diego modern well water prevalence with depth. For details on symbology see the paragraph at the beginning of Supplementary Note 1.

Supplementary Fig. 90. Upper Santa Ana Basin modern well water prevalence with depth.
For details on symbology see the paragraph at the beginning of Supplementary Note 1.

Supplementary Fig. 91. Utah Lake Valley modern well water prevalence with depth. For details on symbology see the paragraph at the beginning of Supplementary Note 1

Supplementary Note 2. Estimating groundwater withdrawals within each study area

We analysed groundwater withdrawal data in order to calculate correlations between the depth to which modern groundwater penetrates and groundwater withdrawals.

We examined the spatial distribution of groundwater withdrawals reported by the USGS for 2015 (see Dieter, C.A., Maupin, M.A., Caldwell, R.R., Harris, M.A., Ivahnenko, T.I., Lovelace, J.K., Barber, N.L., and Linsey, K.S., 2018, Estimated use of water in the United States in 2015: U.S. Geological Survey Circular 1441, 65 p., data from: <https://www.sciencebase.gov/catalog/item/get/5af3311be4b0da30c1b245d8>; field title: "TO-WGWT").

Supplementary Fig. 92. Groundwater withdrawals in counties estimated by Dieter, C.A., Maupin, M.A., Caldwell, R.R., Harris, M.A., Ivahnenko, T.I., Lovelace, J.K., Barber, N.L., and Linsey, K.S., Estimated use of water in the United States in 2015. US Geological Survey Circular 1441, 65 pp. (2018). Accessed May 23, 2022 via <https://pubs.er.usgs.gov/publication/cir1441>. The spatial distribution of county-scale groundwater withdrawals expressed in millions of gallons per day highlights the wide variation in groundwater withdrawals across the US. However, because some counties are larger than others (e.g., see large counties in southern California), comparing the groundwater withdrawals across counties is challenging.

The raw groundwater withdrawal data (figure above) are reported as a volumetric flux (in units of millions of gallons per day); these units are imperfect for comparison across counties, because some have larger areas than others. For example, the counties in southern California are substantially larger than those in Kentucky, meaning the volumetric fluxes (i.e., groundwater withdrawals in figure on previous page) cannot be straightforwardly compared in their current units. Further, groundwater use within counties is unevenly distributed; for example, some of the expansive counties in the southern portion of California's Central Valley extend from the Sierra Nevada mountains (where groundwater pumping is limited) across the Tulare Basin of the Central Valley (where groundwater extractions can be high). These limitations with county-scale data led us to apply a downscaling method in order to estimate groundwater withdrawals at a finer spatial resolution.

We downscaled these data using two finer-resolution data products: (i) gridded population density data from <https://sedac.ciesin.columbia.edu/data/set/gpw-v4-population-density-rev11/data-download> (1 km² data population density data for the year 2015; the year 2015 was selected to match the water use data, which are also for the year 2015), and (ii) irrigated land areas from <https://www.sciencebase.gov/catalog/item/5db08e84e4b0b0c58b56e04f> (1 km² irrigation data for the year 2017). Specifically, to downscale the spatial distribution of groundwater withdrawals across the US we completed two steps.

- (i) First, we downscaled county scale irrigation groundwater withdrawals using MODIS irrigated agriculture grids (total county-level irrigation groundwater withdrawals from Dieter et al. (2018), see field entitled: "IR-WGWF"). Specifically, we calculated the total number of 1 km by 1 km grids within each county; next total county-level groundwater irrigation withdrawals were distributed spatially according to the relative intensity of irrigation as estimated in the MODIS-based dataset. In this way, the estimated USGS county-scale groundwater withdrawals were assumed to be spatially distributed according to the MODIS-based irrigation data. This approach introduces uncertainty in a number of ways, one of which being that our irrigated land data does not provide information pertaining to the water source for irrigation (i.e., groundwater-fed versus surface-water-fed irrigation). However, we found that this downscaling is preferable to the alternative of analysing groundwater use at county scales, as counties are often too expansive to be suitable for locally relevant analyses. For example, our downscaling approach distributes Kern County irrigation withdrawals to the Central Valley (where groundwater-sourced irrigation water withdrawals are known to be high) and correctly identifies that minimal (i.e., at- or near-zero) irrigation occurs in the higher elevations of the Sierra Nevada.

Supplementary Fig. 93. Estimating groundwater withdrawals for irrigated agriculture across the contiguous United States. a) 1 km² resolution MODIS-based (i.e., satellite- and land-use-data-based) irrigated lands from <https://www.sciencebase.gov/catalog/item/5db08e84e4b0b0c58b56e04f>. Irrigation data are for the year 2017 (the closest year to 2015, which is the year for which the groundwater withdrawal data are available for). b) Downscaled irrigation groundwater withdrawals for the year 2015 estimated from county-scale USGS irrigation groundwater withdrawals (from <https://www.sciencebase.gov/catalog/item/get/5af3311be4b0da30c1b245d8>; field title: "IR-WGWF").

(ii) Second, we downscaled all non-irrigation groundwater withdrawals using 1 km² population density data from <https://sedac.ciesin.columbia.edu/data/set/gpw-v4-population-density-rev11/data-download> (1 km² data population density data for the year 2015; the year 2015 was selected to match the water use data, which are also for the year 2015). County-scale non-irrigation groundwater withdrawals were calculated by subtracting irrigation groundwater withdrawals (field title: “IR-WGWFr” in <https://www.sciencebase.gov/catalog/item/get/5af3311be4b0da30c1b245d8>) from total groundwater withdrawals (field title: “TO-WGWTot” <https://www.sciencebase.gov/catalog/item/get/5af3311be4b0da30c1b245d8>). County-scale non-irrigation groundwater withdrawals were assumed to be distributed across each county according to the relative distribution of population within each county. An assumption of a scaling relationship between population density and groundwater pumping has been applied in previous studies of the portions of the contiguous United States (e.g., Ferguson, G., Gleeson, T. Vulnerability of coastal aquifers to groundwater use and climate change. *Nature Climate Change* 2, 342-345 (2012)). Like our approach for analysing the spatial patterns of groundwater-fed irrigation (i.e., (i) above), this approach introduces uncertainty into our analyses, as spatial patterns of population density are not necessarily related to spatial patterns of non-irrigation groundwater withdrawals. The estimated spatial distribution of non-irrigation groundwater withdrawals across the United States are displayed in the figure to follow:

Supplementary Fig. 94. Estimating groundwater withdrawals for purposes other than irrigated agriculture across the contiguous United States. a) Population density from <https://sedac.ciesin.columbia.edu/data/set/gpw-v4-population-density-rev11/data-download>. b) Downscaled non-irrigation groundwater withdrawals for the year 2015 estimated by downscaling county-scale USGS-reported groundwater withdrawal data using population density data displayed in panel a.

Next, for each of our study aquifer systems, we quantified annual groundwater withdrawals for the year 2015 by calculating the sum of all estimated groundwater withdrawals taking place within the boundaries of the aquifer system. The estimated total annual irrigation and non-irrigation groundwater withdrawals taking place within the boundaries of each of our n=74 study areas are shown in the following table.

Supplementary Table 2. Estimated annual groundwater withdrawals in each study aquifer system

Aquifer system	Annual groundwater withdrawals estimated for the year 2015 (cubic kilometers per year)	Area within aquifer system boundaries (km²)	Annual groundwater withdrawals estimated for the year 2015 (mm per year)*
Albuquerque Basin	0.142	7057	20.12
Antelope Valley	0.1122	8139	13.78
Big Bear Valley	0.0045	99	45.34
Bighorn Basin	0.0941	20215	4.66
Black Hills Uplift	0.0316	18915	1.67
Black Warrior River Aquifer System	0.2685	50236	5.35
Boise Valley and Homedale Murphy Area	0.64	4569	140.06
Castle Hayne Aquifer	0.1091	9992	10.92
Catahoula Area	0.184	23067	7.98
Central High Plains	5.115	104407	48.99
Central Mississippi Embayment	16.895	89602	188.55
Central Wabash and Bloomington Ridged Plain	0.1947	25798	7.55
Coachella Valley	0.2339	1797	130.12
Cuyama Valley	0.014	483	28.97
Delmarva Peninsula	0.3698	18139	20.39
Denver Basin	0.1385	18087	7.66
Dougherty Plain and Marianna Lowlands	0.4833	18625	25.95
Eagle Valley	0.0115	179	64.27
Eastern Carrizo-Wilcox	0.3279	57255	5.73
Eastern Dakota Aquifer	0.2167	39327	5.51
Eastern Flatwoods Southshores	0.8125	25502	31.86
Eastern Mississippi Embayment	0.6278	60649	10.35
Eastern Silurian-Devonian Aquifers	0.4564	30217	15.1
Eureka and Eel River and Mad River Plains	0.0687	663	103.7
Garber-Wellington Aquifer	0.0726	8114	8.95
Honey Lake Valley	0.0312	2070	15.07
Houston-Galveston Area	0.6642	29036	22.88
Lafayette Area	1.63	65887	24.73

Aquifer system	Annual groundwater withdrawals estimated for the year 2015 (cubic kilometers per year)	Area within aquifer system boundaries (km²)	Annual groundwater withdrawals estimated for the year 2015 (mm per year)*
Long Island	0.6361	3967	160.36
Los Angeles Basin	0.8256	2097	393.61
Lower Coastal Plain	0.1284	18570	6.91
Lower Santa Ynez Valley	0.016	400	40.03
Maryland Western Shores	0.1647	7008	23.5
Mesilla Valley	0.1843	980	187.97
Michigan Basin	0.4815	62179	7.74
Mississippian-Silurian-Devonian Carbonates	0.6808	87758	7.76
Mojave Basin	0.1017	5974	17.02
Mountain Home Plateau	0.0963	3498	27.53
New Jersey Coastal Plain	0.4955	12541	39.51
North Carolina and Virginia Coastal Plain	0.2901	49130	5.9
Northeast Missouri Carbonates	0.1229	29735	4.13
Northern Cambrian-Ordovician Aquifers	0.3132	20419	15.34
Northern Green River Basin	0.1136	11965	9.5
Northern High Plains	9.413	259022	36.34
Ocala Uplift	0.9025	35703	25.28
Ozark Plateaus Aquifer System	1.003	172532	5.81
Palouse Slope	0.1451	15220	9.53
Pearl and Chattahoochee Aquifer System	0.3231	35620	9.07
Powder River Basin	0.1087	63508	1.71
Sacramento Basin	5.488	16195	338.9
Salinas Valley	0.2515	4317	58.24
Salt Lake Valley	0.1607	1251	128.48
San Joaquin Basin	4.43	14170	312.62
San Luis Valley	0.5581	12662	44.08
San Pedro Basin	0.0299	6717	4.45
Santa Clara-Calleguas Basin	0.1455	852	170.79
Santa Rosa Valley	0.1326	487	272.28
Sea Island	0.5193	22897	22.68
South Park Basin	0.001	2706	0.36
Southern High Plains	4.039	74911	53.92
Southern Hills	0.5163	34222	15.09
Stockton Plateau	0.2408	33280	7.23

Aquifer system	Annual groundwater withdrawals estimated for the year 2015 (cubic kilometers per year)	Area within aquifer system boundaries (km²)	Annual groundwater withdrawals estimated for the year 2015 (mm per year)*
Tifton Upland	0.2336	14671	15.92
Tijuana-San Diego Basin	0.0122	455	26.78
Trinity Aquifer System	0.2892	67113	4.31
Tulare Basin	6.013	20199	297.71
Umatilla Basin and Horse Heaven Hills	0.4004	15371	26.05
Upper Carbonate Aquifer	0.0437	11771	3.71
Upper Santa Ana Basin	0.5326	2146	248.22
Utah Lake Valley	0.1124	383	293.18
Vidalia Upland	0.2219	23846	9.31
Western Cambrian-Ordovician Aquifers	0.6543	59360	11.02
Williston Basin	0.1737	241595	0.72
Yakima Basin	0.2447	9955	24.58

* determined by dividing the value in column 2 (in units of km³/year) by the value in column 3 (in units of km²); next, we multiplied by 10⁶ (to convert from units of km/year into units of mm/year)

The maps on the following pages display the downscaled groundwater withdrawals (for irrigation, for purposes other than irrigation, and total groundwater withdrawals for all purposes (i.e., total annual groundwater withdrawals).

We stress that the realism of the estimated irrigation withdrawals presented in the following maps relies on the accuracy of USGS county-scale groundwater withdrawal estimates. It is possible that USGS county-scale groundwater withdrawal estimates are inaccurate in places. For example, there are counties mapped as having high groundwater irrigation withdrawals where relatively small groundwater withdrawal rates occur.

One such example may be the Imperial Valley, where “100% of the valley’s water supply is imported from the Colorado River via the Imperial Dam and the All-American Canal.” (quoting <https://www.usbr.gov/watersmart/bsp/docs/finalreport/secalifornia/secabasinstudy.pdf>), but the USGS annual irrigation groundwater withdrawal estimate for this county is quite high (Imperial County irrigation groundwater withdrawals (field entitled “IR-WGWFr”) estimated as 1303.78 Mgal/d in this dataset: <https://www.sciencebase.gov/catalog/item/get/5af3311be4b0da30c1b245d8>). This county-scale estimate (1303.78 MGal/d, equivalent to 1,755,000 acre-ft/year) may be an overestimate of actual groundwater withdrawals; a different, local-scale USGS report quotes “Groundwater pumping in the Imperial Valley was estimated to be about 25,000 acre-ft/yr (Tompson and others, 2008)” (quoting <https://pubs.usgs.gov/sir/2015/5102/sir20155102.pdf>). This local-scale estimate is substantially lower than the national-scale, county-level estimate for groundwater withdrawals, though we acknowledge that Imperial County (for which groundwater withdrawals are estimated to be 1,755,000 acre-ft/year in <https://www.sciencebase.gov/catalog/item/get/5af3311be4b0da30c1b245d8>) is more expansive than the Imperial Basin (studied in <https://pubs.usgs.gov/sir/2015/5102/sir20155102.pdf>).

▲ We emphasize that the annual groundwater withdrawal estimates we calculate and analyse via rank regressions for each of our study aquifer systems are imperfect.

Formatted: Font: Times New Roman, English (United Kingdom)

Formatted: Normal

Supplementary Fig. 95. Estimated groundwater withdrawals for irrigation across the contiguous United States. Lighter coloured shades represent lower annual groundwater withdrawals; darker shades represent higher annual groundwater withdrawals. Groundwater withdrawals were estimated by downscaling county-level USGS-estimated groundwater withdrawals (for the year 2015; for data see <https://www.sciencebase.gov/catalog/item/get/5af3311be4b0da30c1b245d8>; field title: "IR-WGWF") using 1 km² resolution MODIS-based (i.e., satellite- and land-use-data-based) irrigated lands from: <https://www.sciencebase.gov/catalog/item/5db08e84e4b0b0c58b56e04f>.

Supplementary Fig. 96. Estimated groundwater withdrawals for purposes other than irrigation across the contiguous United States. Lighter coloured shades represent lower annual groundwater withdrawals; darker shades represent higher annual groundwater withdrawals. Groundwater withdrawals were estimated by downscaling county-level USGS-estimated groundwater withdrawals (for the year 2015; for data see <https://www.sciencebase.gov/catalog/item/get/5af3311be4b0da30c1b245d8>) using population density data (from: <https://sedac.ciesin.columbia.edu/data/set/gpw-v4-population-density-rev11/data-download>).

Supplementary Fig. 97. Estimated groundwater withdrawals for all purposes across the contiguous United States. Lighter coloured shades represent lower annual groundwater withdrawals; darker shades represent higher annual groundwater withdrawals. Groundwater withdrawals were estimated by downscaling county-level USGS-estimated groundwater withdrawals (for the year 2015; for data see <https://www.sciencebase.gov/catalog/item/get/5af3311be4b0da30c1b245d8>) using population density data (from: <https://sedac.ciesin.columbia.edu/data/set/gpw-v4-population-density-rev11/data-download>) and irrigation spatial data (from: <https://www.sciencebase.gov/catalog/item/get/5af3311be4b0da30c1b245d8>).

Supplementary Note 3. Regional hydrogeology and hydrostratigraphy of study aquifer systems

Here we review local- and regional-scale hydrogeologic investigations for each of our aquifer systems to better understand the hydrostratigraphy of each study area. We provide a unique subsection for each hydrogeologic study area examined in our study (see Supplementary Notes 3.1-3.74). Our review focuses on the range of depths to the uppermost confining unit or endogenous bedrock. Our analysis is based on as many as three data sources:

- (i) Analysis of hydrogeologic cross sections created on the basis of a local-scale study (see extent of transparent pink bars on cross sections in subsections to follow)
- (ii) Analysis of USGS wells within the boundaries of the study aquifer where the USGS has defined the well as tapping an unconfined or a confined aquifer* (*note: the USGS data used for this analysis derives from all NWIS wells for which such data are available, even those where tritium data are not available; data downloaded April 22, 2022 from <https://waterdata.usgs.gov/nwis/inventory>)
- (iii) Quotes from local-scale studies that we have reviewed that provide information about the depth to confining units in the study area.

We prioritized use of data source (ii) where sufficient data were available (i.e., if sufficient USGS-defined wells were available to estimate the depth to confined conditions (data source (ii)), we used this estimate instead of data sources (i) and (iii)). Estimated depths to confined conditions for each of our n=74 study areas (i.e., aquifer systems) are summarized in Supplementary Fig. 98 and Supplementary Table 3. Specifically, we review USGS-defined well conditions (i.e., wells in unconfined vs. confined aquifers) for each study area, and we also reconstruct hydrogeologic cross sections for each of our study aquifer systems to better understand the depth to confining units in our study areas. We examine cross sections rather than 3D hydrostratigraphic data because 3D hydrostratigraphic data are unavailable for many of our study aquifers (cf. see digital thickness data for the Mississippi Embayment: <https://pubs.er.usgs.gov/publication/sir20085098>). We examine confining units in the context of the definition from a USGS Scientific Investigations Report: "*a hydrogeologic unit composed of clay or a series of clays that impedes or obstructs groundwater flow*" (italicized text quoted directly from Griffith, J.M. (2006). Hydrogeologic maps and sections of the "400-Foot," "600-Foot," and "800-Foot" Sands of the Baton Rouge Area and adjacent aquifers in east and west Baton Rouge, East and West Feliciana, and Pointe Coupee Parishes, Louisiana. US Geological Survey Scientific Investigations Report 2006-5072, 23 pp. Accessed April 18, 2022 via <https://pubs.usgs.gov/sir/2006/5072/report.pdf>); however, if a local-scale report distinguishes another type of confining unit (e.g., shale) we also include these in our reconstructed cross sections.

For a minority of our study areas we lack sufficient USGS-defined wells (i.e., depth distributions of wells that the USGS has classified as tapping either confined or unconfined conditions), we rely on cross sections to estimate the depth to confined conditions. For these study areas where we rely on cross sections to estimate the depth to confined conditions, our estimate is uncertain, as (a) cross sections may not provide adequate representation of the 3D hydrostratigraphy of the broader aquifer system, as they are necessarily constrained to only a portion of each study area, and (b) some cross sections in the primary

Formatted: Font: Times New Roman, English (United Kingdom)

Formatted: Normal

Formatted: Strong

literature do not differentiate confining beds within a given geologic unit (e.g., see Supplementary Note 3.57 and our review of Fig. 5 by Mashburn et al. (2014) that displays the Central Oklahoma aquifer as a single hydrogeologic unit depicted as outcropping at the land surface with no overlying confining bed in their cross section, but the authors also highlight that: “...even though the Central Oklahoma aquifer extends to land surface with a potentiometric surface below the top of the Central Oklahoma aquifer, the groundwater system acts as a confined system due to laterally extensive interbedded mudstone and large contrasts in vertical hydraulic conductivity.” (italicized text quoted directly from Mashburn, S.L., Ryter, D.W., Neel, C.R., Smith, S.J., Correll, J.S., (2014). Hydrogeology and simulation of ground-water flow in the Central Oklahoma (Garber-Wellington) Aquifer, Oklahoma, 1987 to 2009, and simulation of available water in storage, 2010–2059. US Geological Survey Scientific Investigations Report 2013–5219, 92 pp. Accessed May 14, 2021 from https://pubs.usgs.gov/sir/2013/5219/pdf/sir20135219_v2.0.pdf). We highlight that our estimated depths to confined conditions are thus approximations. These estimated depths to confined conditions are presented in Supplementary Figure 98 on the following page, and tabulated in Supplementary Table 3 in the following pages. For a map of the location of each hydrogeologic cross section see Supplementary Note 8 (and Supplementary Fig. 244).

Supplementary Fig. 98. Map of our estimated depth to confined conditions in each of our n=74 study aquifer systems. Each polygon on the map represents one study area. Light blue colours represent shallower depths to confined conditions; darker blue shades represent deeper depths to confined conditions. Each aquifer system title is shown next to (or atop) the polygon. Depths to confined conditions were estimated on the basis of up to three data sources: (i) USGS-defined well conditions (i.e., wells defined as tapping unconfined versus confined conditions by the USGS), (ii) digitization and evaluation of hydrogeologic cross sections derived from local-scale reports, and (iii) quotes within local-scale reports pertaining to the prevalence of confined conditions. For details see Supplementary Notes 3.1-3.74 on the following pages.

Supplementary Table 3. The depth below which * most wells contain minimal (<25%) modern groundwater in our study aquifer systems (columns 3, 4 and 5; see footnote underneath table for quantitative definitions of “most”) and physiographic, water use and climate conditions within the boundaries of each aquifer system (i.e., four potential explanatory variables are in rightmost 4 columns)

Aquifer system title	Broader aquifer system title	* 60% (m below land surface)	* 70% (m below land surface)	* 80% (m below land surface)	(i) Median depth to top of uppermost confining unit based on cross section**	(ii) Depth below which most wells have been defined by the USGS as tapping a confined aquifer***	(iii) Information from local-scale study pertaining to confined conditions	Estimated depth to confining units for the aquifer system****
Sacramento Basin	California Central Valley	92	114	158	The median depth to granitic and metamorphic rock is >663 meters below land surface; however, the aquifer system transitions to confined conditions at shallower depths, despite the lack of a clear confining unit (see (iii)).	Most (>90%) wells at depths of 90-100 m and at depths exceeding 100 m are defined as tapping a confined aquifer.	"Groundwater is typically unconfined to semi-confined in the shallow aquifer system and confined where deeper aquifers are present." quoting Bureau of Reclamation (EIS/EIR) (2003), Environmental Water Account: Draft Environmental Impact Statement Environmental Impact Report, 317 https://usbr.gov/mp/ewa/docs/v1-draft-enviro-impact-statement-environmental-impact-report.pdf	90-100 m (see (ii) to left)
San Joaquin Basin	California Central Valley	134	163	203	A hydrogeologic cross section presented in Fig. 5 by Page and Balding (1973) does not depict a clear confining unit within the aquifer system.	Most (>80%) wells at depths of 140-160 m and at depths exceeding 140 m are defined as tapping a confined aquifer.	Page and Balding (1973) state (quote directly): "The confined water body occurs in the unconsolidated deposits that underlie the E-clay (fig. 6). The base of the confined water body probably is at the top of the Mehrten Formation, but in terms of use its base is considered to be the base of fresh water..."	140-160 m (see (ii) to left)
Tulare Basin	California Central Valley	126	220	232	A hydrogeologic cross section presented in Plate 1 by Croft (1999) suggests that the aquifer system includes a series of low permeability units, including the "E-clay" (primarily the Corcoran clay unit).	Most (>80%) wells at depths of 160-180 m and wells with depths exceeding 160 m are defined as tapping a confined aquifer.	(Croft (1999) state (quote) "Although the E clay is the principal confining bed in the valley (Davis and other, 1959, p. 87-90; Davis and Poland, 1957, p. 426), water-level data indicate that the A clay also is an effective confining bed throughout most of its extent."	160-180 m (see (ii) to left)
Eastern Carrizo-Wilcox	Carrizo-Wilcox	15	15	20	A hydrogeologic cross section presented in Fig. 2.4 by George (2009) depicts a series of confining units including the relatively shallow Reklaw Formation and the Weches Formation.	Most (>80%) wells at depths of 20-30 m and at depths exceeding 20 m are defined as tapping a confined aquifer.	-	20-30 m (see (ii) to left)
Eagle Valley	Carson River Basin	12	72	152	As we created a cross section based on a georeferenced map of basin fill thickness (by	Only half of the deepest wells in our dataset (depths exceeding 200 m) are defined as tapping a confined	Thiros et al. (2010; see section 4 by J.M. Huntington) state that (quote): "Unconfined to confined conditions	>200 m (see (ii) to left)

Aquifer system title	Broader aquifer system title	* 60% (m below land surface)	* 70% (m below land surface)	* 80% (m below land surface)	(i) Median depth to top of uppermost confining unit based on cross section**	(ii) Depth below which most wells have been defined by the USGS as tapping a confined aquifer***	(iii) Information from local-scale study pertaining to confined conditions	Estimated depth to confining units for the aquifer system****
					Artega, 1986), we did not quantitatively interpret the hydrogeologic cross section (as it was not explicitly based on a cross section from the primary literature).	aquifer; the deepest well in the dataset (381 m) is defined as tapping an unconfined aquifer.	are present in the basin-fill sediments... The degree of confinement varies spatially through the valley due to the clay lenses being discontinuous at different depths." and "Although groundwater exists under both confined and unconfined conditions in Carson Valley, no single confining layer extends across the entire valley... the confining layers occur mainly as scattered, discontinuous clay beds, 30 to 70 ft thick, at a depth of 200 to 300 ft."	
Central Wabash and Bloomington Ridget Plain	Central Lowland Till Plain	7	9	10	A hydrogeologic cross section presented in Fig. 2 by Panno et al. (1994) depicts sedimentary aquifers at shallow depths (including the Mahomet Sand member of the Banner Formation) overlying low permeability layers. In some areas carbonate rocks directly underlie the shallow sedimentary aquifers.	Most (>80%) wells at depths of 30-40 m and at depths exceeding 30 m are defined as tapping a confined aquifer.	:	30-40 m (see (ii) to left)
Palouse Slope	Columbia Plateau Regional Aquifer System	30	30	78	We examined a hydrogeologic cross section generated using a web application available here: https://or.water.usgs.gov/proj/cpr/as/index.html . No clear confining unit is depicted on the cross section.	Only half of the deepest wells in our dataset (depths exceeding 200 m) are defined as tapping a confined aquifer; the deepest well in the dataset (276 m) is defined as tapping an unconfined aquifer.	:	>276 m (see (ii) to left)
Umatilla Basin and Horse Heaven Hills	Columbia Plateau Regional Aquifer System	16	16	62	A hydrogeologic cross section presented in Fig. 3 by Herrera et al. (2017) suggests that the aquifer system does not have a clear confining unit. However, the aquifer system is confined at depth by basalt flow interiors (see (iii) to the right).	All (100%) wells at depths of 60-70 m and at depths exceeding 60 m are defined as tapping a confined aquifer. No wells in our dataset have depths between 25 m and 69 m; all wells (n=2) with depths of shallower than 25 m are classified as unconfined.	Herrera et al. (2017) state that (quote): "The uppermost part of the CRBG is often permeable and unconfined, and has a good hydraulic connection with the overlying alluvial aquifer and, in some cases, streams. Permeable interflow zones at depth are confined by the flow interiors."	60-70 m (see (ii) to left)
Yakima Basin	Columbia Plateau Regional Aquifer System	113	126	130	A hydrogeologic section by Kahle et al. (2011) demonstrates the high degree of topographic and	Three or the four deepest wells in our dataset (depths exceeding 200 m) are defined as tapping a confined	:	280-300 m (see (i) to left)

Aquifer system title	Broader aquifer system title	* 60% (m below land surface)	* 70% (m below land surface)	* 80% (m below land surface)	(i) Median depth to top of uppermost confining unit based on cross section**	(ii) Depth below which most wells have been defined by the USGS as tapping a confined aquifer***	(iii) Information from local-scale study pertaining to confined conditions	Estimated depth to confining units for the aquifer system****
					geologic complexity in the Yakima Basin. Their cross section suggests that the uppermost confining unit is 282 meters below land surface (25th-75th percentile range: 209 m to >529 m meters below land surface)	aquifer; the deepest well in the dataset (291 m, located on the northeast periphery of Grandview) is defined as tapping an unconfined aquifer.		
Stockton Plateau	Edwards-Trinity Aquifer System	130	-	-	A hydrogeologic cross section presented in Fig. 6-13 by Meyer et al. (2012) does not depict a clear confining unit within the aquifer system within the uppermost 190-340 m of the aquifer system (median of pink transparent pink bars suggests a lack of a clear confining unit in the uppermost 249 m below the land surface).	(ii) We analysed wells within the study area that the USGS has defined as either unconfined or confined. Only four wells were available in our dataset; they have depths ranging from 30 m to 87 m, and all are classified as tapping unconfined aquifers.	-	>249 m (see (i) to left)
Trinity Aquifer System	Edwards-Trinity Aquifer System	62	66	73	A hydrogeologic cross section presented in Fig. 6-38 by Bruun et al. (2016) depicts layered sequences of sedimentary aquifers and aquitards. The uppermost Washita and Fredericksburg Formations are shown as aquitards, but may contain permeable portions that serve as aquifers (see (iii) below).	Most (>80%) wells at depths of 90-100 m and at depths exceeding 90 m are defined as tapping a confined aquifer.	Bruun et al. (2016) state (quote) "Sand distribution and thickness largely controls the productivity of the aquifer. The depositional environment in the Cretaceous Period resulted in a layered system of aquifers and aquitards in the northern Trinity Group." Mace et al. (1994) state (quote) "Hydrologic parameters for the Washita and Fredericksburg Groups... were estimated on the basis of rock type. These units are composed of approximately 40 percent shale and 60 percent limestone, as indicated by resistivity well logs..."	90-100 m (see (ii) to left)
Dougherty Plain and Marianna Lowlands	Floridan Aquifer System	76	113	163	A hydrogeologic cross section presented in Plate 8 by Williams and Kuniansky (2016) depicts a shallow confining unit (Hawthorn Group) underlain by carbonate aquifers and confining units.	Most (>80%) wells at depths of 30-40 m and at depths exceeding 30 m are defined as tapping a confined aquifer.	-	30-40 m (see (ii) to left)
Eastern Flatwoods Southshores	Floridan Aquifer System	38	46	61	A hydrogeologic cross section presented in Plate 2 by Williams and Kuniansky (2016) shows	Most (>80%) wells at depths of 100-120 m and at depths exceeding 100 m are defined as tapping a confined	-	100-120 m (see (ii) to left)

Aquifer system title	Broader aquifer system title	* 60% (m below land surface)	* 70% (m below land surface)	* 80% (m below land surface)	(i) Median depth to top of uppermost confining unit based on cross section**	(ii) Depth below which most wells have been defined by the USGS as tapping a confined aquifer***	(iii) Information from local-scale study pertaining to confined conditions	Estimated depth to confining units for the aquifer system****
					undifferentiated sand, silt and clay (~70 m thick) underlain by the Hawthorn Group confining unit.	aquifer.		
Lower Coastal Plain	Floridan Aquifer System	9	34	59	A hydrogeologic cross section presented in Fig. 11 by Aucott (1986) depicts a confining unit at depths of ~80 to ~150 m below the land surface.	Most (>80%) wells at depths of 90-100 m and at depths exceeding 90 m are defined as tapping a confined aquifer.	=	90-100 m (see (ii) to left)
Ocala Uplift	Floridan Aquifer System	103	-	-	A hydrogeologic cross section presented in Plate 2 by Williams and Kuniansky (2016) depicts the Suwannee and Ocala limestone units near the land surface but, in some places, it is overlain by a shallow confining unit (the Hawthorn Group).	Most (>80%) wells at depths of 30-40 m and at depths exceeding 30 m are defined as tapping a confined aquifer.	=	30-40 m (see (ii) to left)
Sea Island	Floridan Aquifer System	6	59	75	A hydrogeologic cross section presented in Plate 1 by Williams and Kuniansky (2016) depicts a shallow layer of undifferentiated sand, silt and clay underlain by a confining unit.	Most (>80%) wells at depths of 30-40 m and at depths exceeding 30 m are defined as tapping a confined aquifer.	=	30-40 m (see (ii) to left)
Tifton Upland	Floridan Aquifer System	9	17	52	A hydrogeologic cross section presented in Plate 8 by Williams and Kuniansky (2016) shows the Surficial Aquifer underlain by a relatively shallow confining unit (Hawthorn Group).	Most (>80%) wells at depths of 40-50 m and at depths exceeding 40 m are defined as tapping a confined aquifer.	=	40-50 m (see (ii) to left)
Vidalia Upland	Floridan Aquifer System	7	9	17	A hydrogeologic cross section presented in Plate 8 by Williams and Kuniansky (2016) depicts undifferentiated sand, silt and clay (less than ~20 m thick) underlain by the Hawthorn Group confining unit.	Most (>80%) wells at depths of 10-20 m and at depths exceeding 10 m are defined as tapping a confined aquifer.	=	10-20 m (see (ii) to left)
Catahoula Area	Gulf Coast Regional Aquifer System	73	79	-	42.8 meters below land surface (25th-75th percentile range: 16-73 meters below land surface)	The available USGS well data (n=3 wells) are insufficient to evaluate the depths at which the aquifer system transitions from unconfined to confined conditions.	"The confining units overlying the upper, middle, and lower Catahoula aquifers are primarily clays within the Catahoula, Pascagoula, and Hattiesburg Formations," quoting Halford, K. J., Barber, N. L. (1995).	40-50 m (see (i) to left)

Aquifer system title	Broader aquifer system title	* 60% (m below land surface)	* 70% (m below land surface)	* 80% (m below land surface)	(i) Median depth to top of uppermost confining unit based on cross section**	(ii) Depth below which most wells have been defined by the USGS as tapping a confined aquifer***	(iii) Information from local-scale study pertaining to confined conditions	Estimated depth to confining units for the aquifer system****
							Analysis of ground-water flow in the Catahoula aquifer system in the vicinity of Laurel and Hattiesburg, Mississippi. U.S. Geological Survey Water-Resources Investigations Report 94-4219, 78 pp. Accessed March 31, 2021 from https://pubs.usgs.gov/wri/1994/4219/report.pdf	
Houston-Galveston Area	Gulf Coast Regional Aquifer System	6	10	16	We analysed wells within the study area that the USGS has defined as either unconfined or confined. Most (>80%) wells at depths of 20-30 m and at depths exceeding 20 m are defined as tapping a confined aquifer.	Most (>80%) wells at depths of 20-30 m and at depths exceeding 20 m are defined as tapping a confined aquifer.	:	20-30 m (see (ii) to left)
Lafayette Area	Gulf Coast Regional Aquifer System	3	15	23	A hydrogeologic cross section presented in Fig. 7 by Vahdat-Aboueshagh et al. (2021) suggests that a low permeability clay-rich unit exists near to the land surface in the study area.	Most (>80%) wells at depths of 20-30 m and at depths exceeding 20 m are defined as tapping a confined aquifer.	:	20-30 m (see (ii) to left)
Southern Hills	Gulf Coast Regional Aquifer System	101	105	178	A hydrogeologic cross section presented in Fig. 4 by USGS (2017) depicts a series of interbedded aquifers and low-permeability units.	Most (>80%) wells at depths of 20-30 m and at depths exceeding 20 m are defined as tapping a confined aquifer.	:	20-30 m (see (ii) to left)
Central High Plains	High Plains	35	59	78	A hydrogeologic cross section presented by Macfarlane (1996) suggests that the uppermost confining unit is 43 meters below the land surface (25th-75th percentile range: 0-97 meters below land surface).	Nearly all wells with depths of less than 300 m are defined as tapping unconfined aquifers. All wells (n=3) at depths of 300-340 m are defined as tapping a confined aquifer (Dakota Sandstone aquifer).	:	300-340 m (see (ii) to left)
Northern High Plains	High Plains	20	35	64	A hydrogeologic cross section presented in Fig. 2 by Hallum et al. (2019) depicts relatively deep depths (>200 m) to the uppermost relatively low-permeability unit.	Nearly all wells with depths of less than 240 m are defined as tapping unconfined aquifers. One well with a depth of 300 m is defined as tapping a confined aquifer (Dakota Sandstone aquifer).	:	>240 m (see (ii) to left)
Southern High Plains	High Plains	89	99	117	A hydrogeologic cross section presented in Fig. 15 by Blandford	Nearly all wells are defined as tapping an unconfined aquifer.	:	>270 m (see (ii) to left)

Aquifer system title	Broader aquifer system title	* 60% (m below land surface)	* 70% (m below land surface)	* 80% (m below land surface)	(i) Median depth to top of uppermost confining unit based on cross section**	(ii) Depth below which most wells have been defined by the USGS as tapping a confined aquifer***	(iii) Information from local-scale study pertaining to confined conditions	Estimated depth to confining units for the aquifer system****
					et al. (2008) suggests that the aquifer system does not depict a clear confining unit within the shallow (<200 m) portion of the aquifer system.	including the deepest well in the dataset (274 m).		
Albuquerque Basin	Middle Rio Grande	29	87	117	A hydrogeologic cross section presented in Plate 2 by Connell (2006) depicts thick (>200 m) poorly consolidated aquifers with a high density of faulting.	Nearly all wells are defined as tapping an unconfined aquifer, including the deepest well in the dataset (631 m).	=	>630 m (see (ii) to left)
San Luis Valley	Middle Rio Grande	107	=	=	A hydrogeologic cross section presented in Fig. 5 by Leonard and Watts (1989) does not depict a clear confining unit within the aquifer system. However, the hydrogeologic cross section in Plate 6 by Repllier et al. (1981) does depict a shallow confining unit ("note: this cross section is incorrectly labelled in the cited publication as "Raton Basin"; we confirmed that this cross section represents the San Luis Valley (not the Raton Basin) via personal communication with a Senior Hydrogeologist at the Colorado Geological Survey on April 18, 2022). We added a confining unit to the cross section (top left of this page) on the basis of	Most (>80%) wells at depths of 140-160 m and at depths exceeding 140 m are defined as tapping a confined aquifer.	=	140-160 m (see (ii) to left)
Central Mississippi Embayment	Mississippi Embayment	33	37	45	A hydrogeologic cross section presented in Figs. 68 and 69 by Renken (1998) suggests that the uppermost confining unit (named the Cook Mountain Formation, in the area where the cross section is located) is typically 75 meters below land surface (median vertical offset of the 20 pink transparent bars in the cross section to the left; the 25th-75th percentile range is 49-94 meters below land surface).	Most (>80%) wells at depths of 70-80 m and at depths exceeding 70 m are defined as tapping a confined aquifer.	=	70-80 m (see (i) and (ii) to left)

Aquifer system title	Broader aquifer system title	* 60% (m below land surface)	* 70% (m below land surface)	* 80% (m below land surface)	(i) Median depth to top of uppermost confining unit based on cross section**	(ii) Depth below which most wells have been defined by the USGS as tapping a confined aquifer***	(iii) Information from local-scale study pertaining to confined conditions	Estimated depth to confining units for the aquifer system****
Eastern Mississippi Embayment	Mississippi Embayment	19	29	58	A hydrogeologic cross section presented in Figs. 68 and 69 by Renken (1998) suggests that the uppermost confining unit (named the Cook Mountain Formation, in the area where the cross section is located) is typically 75 meters below land surface (median vertical offset of the 20 pink transparent bars in the cross section to the left; the 25th-75th percentile range is 49-94 meters below land surface).	Most (>80%) wells at depths of 70-80 m and at depths exceeding 70 m are defined as tapping a confined aquifer.	Renken (1998) state that (following text quoted directly): "The alluvial aquifers consist of gravel and sand deposits of Quaternary age and generally contain ground water under unconfined conditions".	70-80 m (see (ii) to left)
Delmarva Peninsula	North Atlantic Coastal Plain	15	21	29	A hydrogeologic cross section presented in Fig. 21 of Sanford et al. (2012) depicts a thin (less than ~50 m) shallow aquifer underlain by a layered aquifer system consisting of aquitards and aquifers.	Most (>80%) wells at depths of 30-40 m and at depths exceeding 30 m are defined as tapping a confined aquifer.	:	30-40 m (see (ii) to left)
Maryland Western Shores	North Atlantic Coastal Plain	8	26	47	A hydrogeologic cross section presented in Figs. 4-6 by Vroblesky et al. (1991) depicts a shallow geologic formation with both permeable and less-permeable subunits (undivided).	Most (>80%) wells at depths of 30-40 m and at depths exceeding 30 m are defined as tapping a confined aquifer.	:	30-40 m (see (ii) to left)
New Jersey Coastal Plain	North Atlantic Coastal Plain	48	53	58	A hydrogeologic cross section presented in Fig. 10 of Masterson et al. (2013) shows a layered aquifer system including clastic sedimentary aquifers and aquitards of varying thicknesses.	Most (>80%) wells at depths of 80-90 m and at depths exceeding 80 m are defined as tapping a confined aquifer.	:	80-90 m (see (ii) to left)
North Carolina and Virginia Coastal Plain	North Atlantic Coastal Plain	5	8	13	A hydrogeologic cross section presented in Plate 9 by Winner Jr. and Coble (1989) depicts a shallow surficial aquifer underlain by a confining unit at depths of less than ~50 m below the land surface.	Most (>80%) wells at depths of 30-40 m and at depths exceeding 30 m are defined as tapping a confined aquifer.	:	30-40 m (see (ii) to left)
Powder River Basin	Northern Great Plains	4	16	30	A hydrogeologic cross section presented in Fig. 4 by Long et al. (2018) depicts relatively shallow confining units in some areas.	Most (>80%) wells at depths of 60-70 m and at depths exceeding 60 m are defined as tapping a confined aquifer.	:	60-70 m (see (ii) to left)

Aquifer system title	Broader aquifer system title	* 60% (m below land surface)	* 70% (m below land surface)	* 80% (m below land surface)	(i) Median depth to top of uppermost confining unit based on cross section**	(ii) Depth below which most wells have been defined by the USGS as tapping a confined aquifer***	(iii) Information from local-scale study pertaining to confined conditions	Estimated depth to confining units for the aquifer system****
					and deeper depths to the uppermost confining unit in other areas.			
Williston Basin	Northern Great Plains	7	16	28	A hydrogeologic cross section presented in Fig. 4 by Long et al. (2018) depicts complex layered sedimentary sequences, with relatively deep depths to the uppermost confining unit in the northeast portion of the hydrogeologic study area.	Most (>80%) wells at depths of 140-160 m and at depths exceeding 140 m are defined as tapping a confined aquifer.	:	140-160 m (see (ii) to left)
Eastern Silurian-Devonian Aquifers	Northern Midwest Aquifer System	14	16	29	A hydrogeologic cross section presented in Fig. 20 by Young (1992) shows a drift aquifer atop the Silurian-Devonian carbonates, underlain by the Maquoketa confining unit.	Most (>80%) wells at depths of 30-40 m and at depths exceeding 30 m are defined as tapping a confined aquifer.	:	30-40 m (see (ii) to left)
Mississippian-Silurian-Devonian Carbonates	Northern Midwest Aquifer System	14	32	77	A hydrogeologic cross section presented in Fig. 20 by Young (1992) shows Pleistocene deposits overlying Mississippian- to Silurian-aged carbonate rocks atop a series of confining units.	Most (>80%) wells at depths of 60-70 m and at depths exceeding 60 m are defined as tapping a confined aquifer.	:	60-70 m (see (ii) to left)
Northeast Missouri Carbonates	Northern Midwest Aquifer System	41	70	:	A hydrogeologic cross section presented in Plate 1 by Young (1992) depicts layered clastic and carbonate sedimentary units with relatively shallow confining units in most of the area.	Most (>80%) wells at depths of 50-60 m and at depths exceeding 50 m are defined as tapping a confined aquifer.	:	50-60 m (see (ii) to left)
Northern Cambrian-Ordovician Aquifers	Northern Midwest Aquifer System	59	73	105	A hydrogeologic cross section in Plate 1 by Young (1992) depicts sedimentary deposits overlying Precambrian bedrock.	Most (>80%) wells at depths of 160-180 m and at depths exceeding 160 m are defined as tapping a confined aquifer.	:	160-180 m (see (ii) to left)
Upper Carbonate Aquifer	Northern Midwest Aquifer System	93	134	140	A hydrogeologic cross section presented in Fig. 4 by Savoca et al. (1999) depicts Quaternary deposits (~50 m thick) overlying a carbonate rock aquifer (~100-150 m thick).	Most (>80%) wells at depths of 20-30 m and at depths exceeding 20 m are defined as tapping a confined aquifer.	Savoca et al. (1999) write (quote) "The Upper carbonate aquifer underlies the northern part of the study area and consists of 250 to 600 ft of Ordovician- and Devonian-age limestone, dolomite, dolomitic limestone, and shale (table 1; fig. 4, hydrogeologic section A-A') of shallow marine origin. The aquifer is	20-30 m (see (ii) to left)

Aquifer system title	Broader aquifer system title	* 60% (m below land surface)	* 70% (m below land surface)	* 80% (m below land surface)	(i) Median depth to top of uppermost confining unit based on cross section**	(ii) Depth below which most wells have been defined by the USGS as tapping a confined aquifer***	(iii) Information from local-scale study pertaining to confined conditions	Estimated depth to confining units for the aquifer system****
							overlain by unconsolidated Quaternary- and Cretaceous-age deposits (sand, gravel, and clay) and is unconfined except in areas where overlying fine-grained deposits produce locally confined conditions"	
Western Cambrian-Ordovician Aquifers	Northern Midwest Aquifer System	56	87	103	A hydrogeologic cross section presented in Fig. 5 by Seaberg (2000) depicts layered sedimentary sequences including discontinuous aquitards (Glenwood-Platteville-Decorah units) and more continuous aquitards (e.g., St. Lawrence Confining Bed).	Most (>80%) wells at depths of 30-40 m and at depths exceeding 30 m are defined as tapping a confined aquifer.	:	30-40 m (see (ii) to left)
Mesilla Valley	Rincon-Mesilla Valleys	37	64	93	A hydrogeologic cross section presented in Fig. 3 by Robertson et al. (2022) does not depict a clear confining unit within the shallow portion of the aquifer system.	Most (>80%) wells at depths of 240-260 m and at depths exceeding 240 m are defined as tapping a confined aquifer.	:	240-260 m (see (ii) to left)
Lower Santa Ynez Valley	Santa Ynez Valley	61	65	71	A hydrogeologic cross section presented in Fig. 2a, 2.5a by the Western Management Area Groundwater Sustainability Agency (2022) suggests that the uppermost confining unit is typically 202 meters below land surface (25th-75th percentile range: 56-243 meters below land surface).	All n=4 wells (with depths ranging from 49 m to 117 m) are classified as unconfined. The available USGS well data are insufficient to evaluate the depths at which the aquifer system transitions from unconfined to confined conditions.	Regarding confined conditions, the Western Management Area Groundwater Sustainability Agency (2022) states (quote) "The main zone throughout most of the Lompoc Plain subarea is separated from the middle zone by lenses of silt and clay that result in confined or partially confined conditions in the main zone. However, in the eastern, southern, and northern portions of the Lompoc Plain subarea, the confining deposits are less continuous or absent...."	200-220 m (see (i) and (iii) to left)
Boise Valley and Homedale Murphy Area	Western Snake River Plain	98	191	-	A hydrogeologic cross section presented in Fig. 39 by Lindholm (1996) suggests that the upper unit (primarily sand and gravel) is underlain by a confining unit ("Middle unit").	Two of the three deepest wells in the dataset (depths range from 209 m to 372 m) as classified as unconfined; the deepest well (372 m deep) is classified as confined. The available USGS well data are insufficient to evaluate the depths at which the aquifer system transitions from	:	>260 m (see (ii) to left)

Aquifer system title	Broader aquifer system title	* 60% (m below land surface)	* 70% (m below land surface)	* 80% (m below land surface)	(i) Median depth to top of uppermost confining unit based on cross section**	(ii) Depth below which most wells have been defined by the USGS as tapping a confined aquifer***	(iii) Information from local-scale study pertaining to confined conditions	Estimated depth to confining units for the aquifer system****
						unconfined to confined conditions.		
Mountain Home Plateau	Western Snake River Plain	127	131	139	A hydrogeologic cross section presented in Fig. 4 by Norton et al. (1982) for the Mountain Home Plateau does not depict a clear confining unit. The median depth to a confined unit exceeds 336 meters below land surface (median of pink transparent bars in cross section; the 25th-75th percentile range is >336 m to >351 m).	There is only n=1 well (with a depth of 112 m) that the USGS has defined as tapping either a confined or unconfined aquifer; this single data point is insufficient to evaluate the depths at which the aquifer system transitions from unconfined to confined conditions.	=	>336 m (see (ii) to left)
Antelope Valley	=	2	18	84	A hydrogeologic cross section presented in Fig. 3 by Duell Jr. (1987) suggests that shallow unconsolidated deposits are underlain by a confining unit comprised of lacustrine clay.	Most (>80%) wells at depths of 120-140 m and at depths exceeding 120 m are defined as tapping a confined aquifer.	=	120-140 m (see (ii) to left)
Big Bear Valley	=	122	163	=	282 meters below land surface (25th-75th percentile range: 121-336 meters below land surface)	There are no USGS wells within the study area that have been defined as tapping an aquifer that is either unconfined or confined.	=	280-300 m (see (i) to left)
Bighorn Basin	=	61	61	69	A hydrogeologic cross section presented in Plate VI by Tauchen et al. (2012) suggests that the aquifer system does not have a single and continuous confining unit at shallow depths.	Most (>80%) wells at depths of 70-80 m and at depths exceeding 120 m are defined as tapping a confined aquifer.	=	70-80 m (see (ii) to left)
Black Hills Uplift	=	398	545	666	A hydrogeologic cross section presented in Fig. 25 by Driscoll et al. (2002) depicts a series of sedimentary rock units dipping to the east, with numerous confining units including the Spearfish confining unit and the Opeche confining unit.	Most (>80%) wells at depths of 180-200 m and at depths exceeding 180 m are defined as tapping a confined aquifer.	=	180-200 m (see (ii) to left)
Black Warrior River Aquifer System (Eutaw and McShan Formations and	=	5	15	40	A hydrogeologic cross section presented in Fig. 3 by Strom and Mallory (1995) suggests that the shallowest confining unit is typically 3.5 m below the land surface, but with wide variability	Most (>80%) wells at depths of 0-10 m and at depths exceeding 0 m are defined as tapping a confined aquifer.	=	0-10 m (see (ii) to left)

Aquifer system title	Broader aquifer system title	* 60% (m below land surface)	* 70% (m below land surface)	* 80% (m below land surface)	(i) Median depth to top of uppermost confining unit based on cross section**	(ii) Depth below which most wells have been defined by the USGS as tapping a confined aquifer***	(iii) Information from local-scale study pertaining to confined conditions	Estimated depth to confining units for the aquifer system****
Tuscaloosa Group					along the cross section (25th-75th percentile range of the depth to uppermost confining unit is 0m to 255 m).			
Castle Hayne Aquifer	=	24	34	49	A hydrogeologic cross section presented in Plate 6 by Winner Jr. and Coble (1989) depicts shallow low-permeability units including the Castle Hayne Confining Unit, Beaufort Confining Unit, and Peedee Confining Unit.	Most (>80%) wells at depths of 10-20 m and at depths exceeding 10 m are defined as tapping a confined aquifer.	=	10-20 m (see (ii) to left)
Coachella Valley	=	232	277	323	A hydrogeologic cross section presented in Fig. 3-3 by MWH (2014) suggests that a relatively continuous aquitard exists in the southeastern portion of the Coachella Valley, but that this unit is absent in the northwestern portion of the area.	(ii) We analysed wells within the study area that the USGS has defined as either unconfined or confined. Nearly all (11 of 12) wells in the Coachella Valley are defined as unconfined, including the deepest wells in the dataset (depths of 323 m and 341 m).	=	>341 m (see (ii) to left)
Cuvama Valley	=	67	189	=	The median depth to a confined unit or endogenous bedrock is >722 m.	The available USGS well data (n=2 wells with depths of 71 m and 183 m) are insufficient to evaluate the depths at which the aquifer system transitions from unconfined to confined conditions.	"The aquifer is considered to be continuous and unconfined with the exception of locally perched aquifers resulting from clays in the formations."	>722 m (see (i) and (iii) to left)
Denver Basin	=	34	46	60	A hydrogeologic cross section presented in Fig. 3 by Malenda and Penn (2020) suggests the top of the uppermost confining unit (the Laramie Formation) is typically 297 meters below the land surface.	Most (>80%) wells at depths of 160-180 m and at depths exceeding 160 m are defined as tapping a confined aquifer.	Malenda and Penn (2020) highlight that confined conditions can exist in the Denver Formation, which overlies the Laramie Formation; they state that "The three Denver aquifer wells represent confined aquifer conditions". Therefore, we do not rely solely on the depth to the Laramie Formation to estimate the depth to confined conditions, instead drawing information from the wells where the USGS has defined whether the well taps unconfined versus confined conditions.	160-180 m (see (ii) to left)
Eastern Dakota Aquifer	=	12	21	30	A hydrogeologic cross section presented in the figure on page 27 by Prior et al. (2003) depicts	Most (>80%) wells at depths of 60-70 m and at depths exceeding 60 m are defined as tapping a confined	=	60-70 m (see (ii) to left)

Aquifer system title	Broader aquifer system title	* 60% (m below land surface)	* 70% (m below land surface)	* 80% (m below land surface)	(i) Median depth to top of uppermost confining unit based on cross section**	(ii) Depth below which most wells have been defined by the USGS as tapping a confined aquifer***	(iii) Information from local-scale study pertaining to confined conditions	Estimated depth to confining units for the aquifer system****
					relatively thick unconsolidated deposits overlying the Dakota Aquifer, which itself overlies dipping sedimentary sequences including several aquitards (e.g., the Upper and Basal Devonian Aquitards).	aquifer.		
Eureka and Eel River and Mad River Plains	=	59	66	=	605 meters below land surface (25th-75th percentile range: 421-767 meters below land surface)	There are no USGS wells within the study area that have been defined as tapping an aquifer that is either unconfined or confined.	"The two younger units, the Scotia Bluffs Sandstone and Carlotta Formation, consist dominantly of coarse-grained clastic sediments of marginal marine deposition and may be important aquifers. Water found in these two formations in the Eureka area is generally confined by interbeds of silt and clay of low permeability or by the over-lying, fine-grained sediment in the Hookton Formation" quoting Johnson, M. J. (1975). Ground-water conditions in the Eureka Area, Humboldt County, California. U.S. Geological Survey Water-Resources Investigations 78-127. 51 pp. Accessed March 20, 2021 from https://pubs.usgs.gov/wri/1978/0127/report.pdf	70-80 m^x (see (iii) to left)
Garber-Wallington Aquifer	=	37	49	60	A hydrogeologic cross section presented in Fig. 5 by Mashburn et al. (2014) does not depict a clear confining unit in the central portion of the study area (but see quote in (ii) to the right).	Most (>80%) wells at depths of 80-90 m and at depths exceeding 80 m are defined as tapping a confined aquifer.	Mashburn et al. (2014) state that (following text quoted directly): "...even though the Central Oklahoma aquifer extends to land surface with a potentiometric surface below the top of the Central Oklahoma aquifer, the groundwater system acts as a confined system due to laterally extensive interbedded mudstone and large contrasts in vertical hydraulic conductivity."	80-90 m (see (ii) to left)
Honey Lake Valley	=	31	75	=	(i) A hydrogeologic cross section presented in Fig. 2 by Mayo et al. (2010) suggests that the unconfined portion of Honey Lake Valley extends to a typical depth of greater than 683 m meters below land surface	There are no USGS wells within the study area that have been defined as tapping an aquifer that is either unconfined or confined.	"Three groundwater systems have been identified in Honey Lake basin: (1) shallow unconfined and semiconfined (<200 m below ground surface (bgs)), (2) deep confined (>200 m bgs), and (3) geothermal" quoting Mayo, A. L., Henderson, R.	200-220 m based on (iii)

Aquifer system title	Broader aquifer system title	* 60% (m below land surface)	* 70% (m below land surface)	* 80% (m below land surface)	(i) Median depth to top of uppermost confining unit based on cross section**	(ii) Depth below which most wells have been defined by the USGS as tapping a confined aquifer***	(iii) Information from local-scale study pertaining to confined conditions	Estimated depth to confining units for the aquifer system****
					(median of pink bars shown in cross section to the left).		M., Tingey, D., Webber, W. (2010). Chemical evolution of shallow playa groundwater in response to post-pluvial isostatic rebound, Honey Lake Basin, California–Nevada, USA. Hydrogeology Journal , 18, 725-747.	
Long Island	:	101	138	149	A hydrogeologic cross section presented by Smolensky et al. (1990) suggests that the aquifer system has two primary confining units, the Gardiners Clay and the Raritan Confining Unit.	Most (>80%) wells at depths of 120-140 m and at depths exceeding 120 m are defined as tapping a confined aquifer.	:	120-140 m (see (ii) to left)
Los Angeles Basin	:	160	346	397	A hydrogeologic cross section presented in Fig. 4 by Reichard et al. (2003) does not depict a continuous confining unit in the study area.	We could not identify a depth where most (>80%) wells with the depth range and at deeper depths are defined as tapping a confined aquifer. Three of the four deepest wells in our dataset (depths of 396 m, 399 m, 445 m, and 472 m) are classified as unconfined.	:	>472 m (see (ii) to left)
Michigan Basin	:	89	:	:	A hydrogeologic cross section presented in Fig. 7 by Westjohn et al. (1998) depicts a series of stacked confining units underlying relatively thick (>100 m) glacial deposits.	Most (>80%) wells at depths of 40-50 m and at depths exceeding 40 m are defined as tapping a confined aquifer.	:	40-50 m (see (ii) to left)
Mojave Basin	:	1	31	60	A hydrogeologic cross section presented in Fig. 2a by Kulongoski et al. (2003) does not depict a clear confining unit within the aquifer system. The median depth to basement igneous and metamorphic rocks as depicted in the cross section is >512 m.	We could not identify a depth where most (>80%) wells with the depth range and at deeper depths are defined as tapping a confined aquifer. The two deepest wells in our dataset (depths of 228 m and 314 m) are classified as unconfined.	Kulongoski et al. (2003) highlight that unconfined conditions prevail in parts of the aquifer system; they state (following text quoted directly): "These deposits (QTa) consist of unconsolidated to moderately consolidated gravel, sand, silt, and clay deposited in the Pleistocene and late Pliocene, and overlie a crystalline complex of igneous and metamorphic rocks (pTb)." (California Dept. Of Water Resources, 1967). Waters from this mostly unconfined aquifer..."	>512 m (see (ii) to left)
Northern Green River	:	27	49	82	Fig. 7 by Bartos et al. (2015) suggests that the top if the	The available USGS well data are insufficient to evaluate the depths at	Bartos et al. (2015) state, with regards to the presence of confining	180-200 m (see (i) to left)

Aquifer system title	Broader aquifer system title	* 60% (m below land surface)	* 70% (m below land surface)	* 80% (m below land surface)	(i) Median depth to top of uppermost confining unit based on cross section**	(ii) Depth below which most wells have been defined by the USGS as tapping a confined aquifer***	(iii) Information from local-scale study pertaining to confined conditions	Estimated depth to confining units for the aquifer system****
Basin					uppermost confining unit is typically 193 meters below land surface (i.e., the median length of the pink transparent bars in cross section; the lower-upper quartile range is 137-225 meters).	which the aquifer system transitions from unconfined to confined conditions. Wells with depths of 335 m and 416 m are classified as confined, where the shallowest wells (all with depths of less than 26 m) are classified as unconfined.	units, that (quote) "Confining units are the Wilkins Peak and Tipton confining units, where impermeable, the Laney Member of the Green River Formation also is a confining unit." As the Laney Formation overlies the Wilkins Formation, it is possible that confined conditions prevail at shallower depths than the top of the Wilkins Formation. However, without spatial data defining low-permeability parts of the Laney Formation, we cannot shorten the length of the pink bars depicted on the cross section such that their bottoms are shallower than the top of the Wilkins Peak Formation. Our estimated depth to confined conditions may overestimate actual depths to confined conditions in some areas.	
Ozark Plateaus Aquifer System	=	66	113	192	A hydrogeologic cross section presented in Fig. 3 by Clark et al. (2019) shows thick sequences of carbonate rock with interbedded confining units. The St. Francois confining unit exists at deep depths (>300 m) and, in the northwest, a shallower and thin confining unit exists (the Ozark confining unit).	Most (>80%) wells at depths of 940-960 m and at depths exceeding 940 m are defined as tapping a confined aquifer.	=	940-960 m (see (ii) to left)
Pearl and Chattahoochee Aquifer System	=	38	55	128	(i) A hydrogeologic cross section presented in Fig. 3 by Kidd (1987) depicts a series of dipping sedimentary formations. Note that the cross section presented by Kidd (1987) lacks a quantitative vertical axis; whereas the cross sections we consulted for the other n=73 aquifer systems have quantitative vertical axes. We scaled the vertical scale of the cross section by Kidd (1987) by matching the topography depicted in the figure with an elevation profile along the cross section path. We stress	Most (>80%) wells at depths of 20-30 m and at depths exceeding 20 m are defined as tapping a confined aquifer.	=	20-30 m (see (ii) to left)

Aquifer system title	Broader aquifer system title	* 60% (m below land surface)	* 70% (m below land surface)	* 80% (m below land surface)	(i) Median depth to top of uppermost confining unit based on cross section**	(ii) Depth below which most wells have been defined by the USGS as tapping a confined aquifer***	(iii) Information from local-scale study pertaining to confined conditions	Estimated depth to confining units for the aquifer system****
					that this cross section is more uncertain than others; we did not use the cross section for this study area to estimate the depth to confined conditions (i.e., we use data source (ii) described to the right).			
Salinas Valley	=	89	195	222	A hydrogeologic cross section presented in Fig. 3 by Hall (1992) demonstrates widespread confining beds (e.g., the Salinas Valley Aquitard in the northwest of the valley). The median depth to the top of the uppermost confined unit is 31 meters (25th-75th percentile range is 9 meters to 49 meters below land surface; see pink transparent bars in cross section to the left).	There are no USGS wells within the study area that have been defined as tapping an aquifer that is either unconfined or confined.	Hall (1992) states, with respect to confining conditions, (quote), "The blue clay, overlying the 180 foot aquifer, ranges from 25 feet thick at Salinas to more than 100 feet thick at Nashua Road. Is it known as the Salinas Aquitard and is composed mostly of blue marine clays with some silts". A report by Harding ESE (2001) on the hydrostratigraphy area states that (quote), "The lateral extent of the 180-Foot Aquifer is generally defined by the overlying [Salinas Valley Aquitard] clay, which maintains confined conditions throughout much of the study area." Vengosh et al. (2002) state that; "The Salinas Valley has a relatively deep, confined "400-foot" aquifer, overlain by a "180-foot" aquifer and a shallower perched aquifer, all made up of alluvial sand, gravel and clay deposits".	30-40 m (see (i) to left)
Salt Lake Valley	=	294	=	=	A hydrogeologic cross section presented in Fig. 4 by Manning and Solomon (2005) labels a fine-grained confining layer at shallow depths (<100 m) across much of the study area.	Most (>80%) wells at depths of 50-60 m and at depths exceeding 50 m are defined as tapping a confined aquifer.	=	50-60 m (see (ii) to left)
San Pedro Basin	=	201	244	280	A hydrogeologic cross section presented in Fig. 2 by Hopkins et al. (2014) depicts a clay layer in the central portion of the basin.	Most (>80%) wells at depths of 300-320 m and at depths exceeding 300 m are defined as tapping a confined aquifer.	=	300-320 m (see (ii) to left)
Santa Clara-Calleguas Basin	=	77	102	226	A hydrogeologic cross section presented in Fig. 8 by Hanson et al. (2002) depicts a multi-layered aquifer system with a confining	Most (>80%) wells at depths of 50-60 m and at depths exceeding 50 m are defined as tapping a confined aquifer.	=	50-60 m (see (ii) to left)

Aquifer system title	Broader aquifer system title	* 60% (m below land surface)	* 70% (m below land surface)	* 80% (m below land surface)	(i) Median depth to top of uppermost confining unit based on cross section**	(ii) Depth below which most wells have been defined by the USGS as tapping a confined aquifer***	(iii) Information from local-scale study pertaining to confined conditions	Estimated depth to confining units for the aquifer system****
					unit relatively close to the coast.			
Santa Rosa Valley	=	31	70	209	In the hydrogeologic cross section in Fig. 2-9 by Santa Rosa Plain Advisory Panel (2014), the median depth to the top of the Petaluma Formation—which is depicted as low-permeability by Woolfenden and Nishikawa (2014) in their Fig. 5—is 111 m (25th-75th percentile range of the depth to the top of the Petaluma Formation is 68 to 132 meters below land surface).	There are no USGS wells within the study area that have been defined as tapping an aquifer that is either unconfined or confined.	Santa Rosa Plain Advisory Panel (2014) state that (following text quoted directly): "In most parts of the study area, shallow groundwater flow is unconfined, but, at depth, groundwater flow is confined." Further, Woolfenden and Nishikawa (2014) states (quote): "The Petaluma Formation is dominated by fine-grained materials, either in thick beds or as interstitial material in poorly sorted silty and clayey sands or gravels." (see also their conceptual model in Fig. 5 by Woolfenden and Nishikawa (2014) depicting the Petaluma Formation as a low-permeability formation).	100-120 m (see (i) to left)
South Park Basin	=	5	12	67	A hydrogeologic cross section presented in Fig. 11 by Barkmann et al. (2013) demonstrates the complexity of the sedimentary rock sequences in the South Park Basin. Among the 20 evenly spaced pink transparent bars (the length of each representing the depth to the uppermost confining unit), the median depth to a confining unit is >576 meters below land surface (25th-75th percentile range: 224 m to >619 m).	All wells (n=52) in the South Park Basin (depths of 0.15 m to 183 m) are classified as unconfined.	=	>576 m (based on (i), consistent with data presented in (ii))
Tijuana-San Diego Basin	=	8	8	37	A hydrogeologic cross section presented in a poster by Anders et al. (2012) depicts a series of sedimentary sequences including shallow Quaternary-aged alluvium underlain by Pliocene- to Eocene-aged formations.	Most (>80%) wells at depths of 140-160 m and at depths exceeding 140 m are defined as tapping a confined aquifer.	=	140-160 m (see (ii) to left)
Upper Santa Ana Basin	=	330	348	351	A hydrogeologic cross section presented in Fig. 2-11 by Wildermuth (2005) does not depict a clear continuous confining unit in the Upper Santa	We could not identify a depth where most (>80%) wells with the depth range and at deeper depths are defined as tapping a confined aquifer. The three deepest wells in	=	>360 m (see (ii) to left)

Aquifer system title	Broader aquifer system title	* 60% (m below land surface)	* 70% (m below land surface)	* 80% (m below land surface)	(i) Median depth to top of uppermost confining unit based on cross section**	(ii) Depth below which most wells have been defined by the USGS as tapping a confined aquifer***	(iii) Information from local-scale study pertaining to confined conditions	Estimated depth to confining units for the aquifer system****
					Ana Basin.	our dataset (depths of 330 m, 338 m, 360 m) are classified as unconfined.		
Utah Lake Valley	-	276	-	-	A hydrogeologic cross section presented in Fig. 47 by Cederberg et al. (2009) depicts a series of shallow (i.e., near surface) confining units.	Most (>80%) wells at depths of 10-20 m and at depths exceeding 10 m are defined as tapping a confined aquifer.	-	10-20 m (see (ii) to left)

* the depth below which most samples contain minimal (<25%) modern water, where the value defining “most” is reported in the column headings (columns 3, 4 or 5) as one of (i) 60%, (ii) 70%, or (iii) 80%

Formatted: Font: Not Bold

~~We tested** determined as the robustness median vertical offset between the land surface and the top of the Spearman-rank correlation coefficients presented in the main text by removing and re-running our correlation analyses after removing the N=17 aquifer systems where uppermost confining unit or endogenous bedrock as depicted on the depth below which ever 60%, 70% or 80% cross section (i.e., the median vertical length (see y-axis for scale) of deeper the 20 transparent pink bars on the cross section)~~

~~*** determined on the basis of depth profiles of wells contain minimal modern groundwater (<25%) appears that the USGS has categorized as tapping an imperfect unconfined (or confined) aquifer~~

~~**** determined on the basis of the combined available from information sources (i), (ii) and (iii) (see columns to left); if a specific information source (i.e., categories (i), (ii) or (iii) in the adjacent columns) was used to generate the estimate the roman numeral specifying that information source is shown here~~

^x the uncertainty in our estimated depth to confined conditions for the Eureka and Mad River Plains aquifer system is substantial. The depth to confined conditions was approximated as roughly half the maximum thickness of the Hookton Formation on the basis of descriptions by Johnson, M. J. (1975). Ground-water conditions in the Eureka Area, Humboldt County, California. US Geological Survey Water-Resources Investigations 78-127, 51 pp. Accessed March 20, 2021 from <https://pubs.usgs.gov/wri/1978/0127/report.pdf>

Formatted: Font: Times New Roman, English (United Kingdom)

Formatted: Font: Times New Roman, English (United Kingdom)

Formatted: Font: Times New Roman, English (United Kingdom)

Formatted: Font: Times New Roman, English (United Kingdom)

Formatted: Font: Times New Roman, English (United Kingdom)

Formatted: Font: Times New Roman, English (United Kingdom)

Formatted: Font: Times New Roman, English (United Kingdom)

3.1 Sacramento Basin, California Central Valley

Supplementary Fig. 99. Hydrogeologic cross section. 20 equally spaced transparent pink bars overlie the cross section; each shaded bar depicts the vertical offset from the land surface to the top of the uppermost confining unit or endogenous bedrock.

Supplementary Fig. 100. Vertical variations in the prevalence of wells that have been defined as tapping an unconfined or a confined aquifer by the USGS. The smaller squares represent 10 m depth intervals from the land surface to 100 m; the larger squares represent 20 m intervals from 100 m to 300 m below the land surface.

The Sacramento Basin is located in north-central California.

(i) A hydrogeologic cross section presented in Fig. 6-6 by Bureau of Reclamation (2003) does not depict a clear confining unit. The median depth to granitic and metamorphic rock is >663 meters below land surface; however, the aquifer system transitions to confined conditions at shallower depths, despite the lack of a clear confining unit (see (iii)).

(ii) We analysed wells within the study area that the USGS has defined as either unconfined or confined. Most (>80%) wells at depths of 90-100 m or at depths exceeding 90 m are defined as tapping a confined aquifer.

(iii) The Bureau of Reclamation (2003) highlights that deeper depths can be confined (direct quote from report: "Groundwater is typically unconfined to semi-confined in the shallow aquifer system and confined where deeper aquifers are present").

Depth to confined conditions: 90-100 m based on (ii) above

Reference: Bureau of Reclamation (EIS/EIR) (2003). Environmental Water Account: Draft Environmental Impact Statement Environmental Impact Report. 317 <https://usbr.gov/mp/ewa/docs/v1-draft-enviro-impact-statement-environmental-impact-report.pdf>

The table below presents a series of published quotes (see quotation marks denoting text quoted from another publication, which is cited following the quotation marks with the full reference written in full below the table). The leftmost column lists a title of a hydrogeologic formation depicted in the cross section on the previous page. The rightmost column presents a quote from a hydrogeological study (see base of table for citation). The quote has been annotated with colored text to highlight how we categorized each layer (i.e., see categories in the center column in the table). Specifically: (i) blue text highlights portions of a quote that provide insights into the degree of consolidation of the formation, (ii) red text highlights portions of a quote that categorize the formation as an aquifer or an aquitard (i.e., higher versus lower permeability in the context of local hydrogeologic formations), and (iii) green text highlights portions of a quote that provide information about the lithology of the formation.

Supplementary Table 4. Hydrostratigraphy details for the Sacramento Valley

Formation name	Category	Quote
Alluvial Fan Deposit	Unconsolidated aquifer	"The alluvial fan is highly suitable for banking purposes, as it generally consists of permeable river deposits with high well yields that allow quick recovery." (EWA, 2003)
Tehama Formation	Unconsolidated aquifer	"The Tehama Formation in the western portion of the basin is derived from Coast Range sediment. In most of the Sacramento Groundwater Basin, the Tuscan, Mehrten, and Tehama formations are overlain with relatively thin alluvial deposits." (EWA, 2003) "The groundwater basin west of the Sacramento River is composed of the Tehama Formation, which has exhibited subsidence in Yolo County" (EWA, 2003) "Of particular importance to this study is the Plio-Pleistocene Tehama Formation, a productive aquifer described in some detail below..." Davisson and Criss (1993) "In the subsurface the Tehama deposits generally are water-saturated and unconsolidated, except near the base of the section where moderately consolidated gravels occur" Davisson and Criss (1993) "The Tehama Formation is a 600-900 m thick fluvial deposit that extends from the Coast Ranges to the axis of the Sacramento Valley, and from Red Bluff in the north to the Montezuma Hills in the eastern Delta region (Olmsted and Davis, 1961). This formation, which has been locally deformed by late Cenozoic uplift and tilting along the western basin margin, consists of detritus derived from the rapidly rising Coast Ranges (Thomasson et al., 1960; Loewen et al., 1992). The deposits are characterized by yellowish or brownish to blue-green clays interbedded with sands and gravels." Davisson and Criss (1993)
Eocene and Post-Eocene Continental Deposit	Clastic sedimentary rock aquifer (consolidated or semi-consolidated rock)	"The base of the post-Eocene continental deposits is equivalent to the base of the Tehama Formation of Pliocene age which in some places at least may be of late Oligocene and early Miocene age. The post-Eocene deposits contain most of the fresh ground water in the valley." Page, (1974). "The continental sediments consist mostly of sand and gravel interbedded and mixed with clay and silt deposited by streams and lakes." Thiros et al. (2010)

Formation name	Category	Quote
Granitic and Metamorphic rocks of the Sierra Nevada	Endogenous bedrock	" Granitic, volcanic, and metamorphic rocks that crop out and underlie the eastern part of the valley form an almost impermeable boundary for the basin-fill groundwater system. " (EWA, 2003)

Page, R. W. (1974). Base and thickness of the post-Eocene continental deposits in the Sacramento Valley, California (No. 45-73). US Geological Survey. <https://pubs.usgs.gov/wri/1973/0045/report.pdf>

Environmental Water Account ("EWA") (2003). Draft Environmental Impact Statement Environmental Impact Report Volume I Chapters 1-9. <https://www.usbr.gov/mp/ewa/docs/v1-draft-enviro-impact-statement-environmental-impact-report.pdf>

Thiros, S. A., Bexfield, L. M., Anning, D. W., & Huntington, J. M. (2010). Conceptual understanding and groundwater quality of selected basin-fill aquifers in the southwestern United States (No. 1781). US Geological Survey. <https://pubs.usgs.gov/pp/1781/>

Marchand, D. E., & Allwardt, A. (1977). Late Cenozoic stratigraphic units, northeastern San Joaquin Valley, California (No. 77-748). US Geological Survey. <https://pubs.usgs.gov/bul/1470/report.pdf>

Davisson, M. L., & Criss, R. E. (1993). Stable isotope imaging of a dynamic groundwater system in the southwestern Sacramento Valley, California, USA. *Journal of Hydrology*, 144, 213-246.

3.2 San Joaquin Basin, California Central Valley

Supplementary Fig. 101. Hydrogeologic cross section. 20 equally spaced transparent pink bars overlie the cross section; each shaded bar depicts the vertical offset from the land surface to the top of the uppermost confining unit or endogenous bedrock.

Supplementary Fig. 102. Vertical variations in the prevalence of wells that have been defined as tapping an unconfined or a confined aquifer by the USGS. The smaller squares represent 10 m depth intervals from the land surface to 100 m; the larger squares represent 20 m intervals from 100 m to 300 m below the land surface.

The San Joaquin Basin is located in central California.

(i) A hydrogeologic cross section presented in Fig. 5 by Page and Balding (1973) does not depict a clear confining unit within the aquifer system.

(ii) We analysed wells within the study area that the USGS has defined as either unconfined or confined. Most (>80%) wells at depths of 140-160 m and at depths exceeding 140 m are defined as tapping a confined aquifer.

(iii) Page and Balding (1973) state (quote directly): “The confined water body occurs in the unconsolidated deposits that underlie the E-clay (fig. 6). The base of the confined water body probably is at the top of the Mehrten Formation, but in terms of use its base is considered to be the base of fresh water...”

Depth to confined conditions: 140-160 m based on (ii) above

Reference: Page, R. W., Balding, G. O. (1973). *Geology and quality of water in the Modesto-Merced area, San Joaquin Valley, California, with a brief section on hydrology.* US Geological Survey Water-Resources Investigations Report 73-6, 77 pp. Accessed April 11, 2022 via <https://pubs.usgs.gov/wri/1973/006/report.pdf>

The table below presents a series of published quotes (see quotation marks denoting text quoted from another publication, which is cited following the quotation marks with the full reference written in full below the table). The leftmost column lists a title of a hydrogeologic formation depicted in the cross section on the previous page. The rightmost column presents a quote from a hydrogeological study (see base of table for citation). The quote has been annotated with colored text to highlight how we categorized each layer (i.e., see categories in the center column in the table). Specifically: (i) blue text highlights portions of a quote that provide insights into the degree of consolidation of the formation, (ii) red text highlights portions of a quote that categorize the formation as an aquifer or an aquitard (i.e., higher versus lower permeability in the context of local hydrogeologic formations), and (iii) green text highlights portions of a quote that provide information about the lithology of the formation.

Supplementary Table 5. Hydrostratigraphy details for the San Joaquin Basin

Formation name	Category	Quote
Qb – Flood-basin deposits (unconsolidated aquitard due to high impermeable clay layer)	Sedimentary rock aquitard (consolidated or semi-consolidated rock)	"They consist of intercalated lenses of bluish-gray, brown, and reddish-brown fine sand, silt, and clay. In the subsurface, the deposits are interbedded with the younger alluvium and probably in part with the older alluvium. " (Page and Balding, 1973). Figure 5 in Page and Balding, 1973, the explanation shows that " Flood-basin deposits " under " Unconsolidated rocks ". "As indicated on drillers' logs, they range in thickness from 0 to about 100 feet. Because of their impermeable clayey nature , the flood-basin deposits would yield only very small quantities of water to wells " (Page and Balding, 1973).
Qya – Younger alluvium	Unconsolidated aquifer	"It consists mostly of fine sand, sand, and gravel with little or no hardpan. " (Page and Balding, 1973). Figure 5 in Page and Balding, 1973, the explanation shows that " Younger alluvium " under " Unconsolidated rocks ". "Because in most places the younger alluvium is not completely saturated, it probably will yield only moderate quantities of water to wells. " (Page and Balding, 1973).
Q1 – lacustrine and marsh deposits (E-clay)	Sedimentary rock aquitard (consolidated or semi-consolidated rock)	"The lacustrine and marsh deposits (E-clay) and the flood-basin deposits yield little water to wells " (Page and Balding, 1973). Table 1 in Page and Balding, 1973, Lithologic characteristic: " Silt, silty clay, and clay, gray and blue " and Water-bearing character: " Confining bed, very small hydraulic conductivities. "
Qca – Older alluvium	Unconsolidated aquifer	"It consists of intercalated beds of gravel, sand, silt, and clay, with some hardpan. " (Page and Balding, 1973). Figure 5 in Page and Balding, 1973, the explanation shows that " Older alluvium " under " Unconsolidated rocks ". "The older alluvium is the most extensively developed aquifer in the Modesto-Merced area, yielding water to large numbers of domestic, irrigation, industrial, and public-supply wells. " (Page and Balding, 1973).
QTc – Continental deposits	Unconsolidated aquifer	"They consist of a gently southwestward-dipping alluvium of poorly sorted gravel, sand, silt, and clay. And they are generally finer grained than the overlying older alluvium (fig. 12)." (Page and Balding, 1973). Figure 5 in Page and Balding, 1973, the explanation shows that " Continental deposits " under " Unconsolidated rocks ". " Yields to wells are as large as 281 cfm (2,100 gpm) and specific capacities as large as 3 ft²/min (22 gpm per foot). " (Page and Badling, 1973).

Formation name	Category	Quote
Tm – Mehrten Formation	Sedimentary rock aquifer (consolidated or semi-consolidated rock)	“The Mehrten Formation, of Miocene and Pliocene age, crops out near the eastern edge of the area (fig. 5). It consists of fluvialite deposits of sandstone, breccia, conglomerate, tuff, siltstone, and claystone (Davis and Hall, 1959, p. 9-10; Piper and others, 1939, p. 61-67).” (Page and Balding, 1973). “The Mehrten is one of the important aquifers in the Modesto-Merced area, and water wells in the eastern part of the area commonly penetrate it (fig. 6).” (Page and Balding, 1973). Figure 5 in page and Balding, 1973, the explanation shows that “Mehrten Formation” under “Consolidated rocks”.
pTb – Basement complex	Endogenous bedrock	“The exposed basement complex consists largely of metasedimentary and 370etavolcanics rocks of pre-Tertiary age (Bateman and others, 1963, pl. 1).” (Page and Badling, 1973). “Where the basement complex occurs at or near the surface, only small quantities of water are yielded to wells through narrow joints and fractures.” (Page and Balding, 1973).

Page, R. W., Balding, G. O. (1973). Geology and quality of water in the Modesto-Merced area, San Joaquin Valley, California, with a brief section on hydrology. (No.6-73). US Geological Survey Water-Resources Investigations Report 73-6. 77 pp. Accessed June 16, 2022 via <https://pubs.er.usgs.gov/publication/wri736>

3.3 Tulare Basin, California Central Valley

Supplementary Fig. 103. Hydrogeologic cross section. 20 equally spaced transparent pink bars overlie the cross section; each shaded bar depicts the vertical offset from the land surface to the top of the uppermost confining unit or endogenous bedrock.

Supplementary Fig. 104. Vertical variations in the prevalence of wells that have been defined as tapping an unconfined or a confined aquifer by the USGS. The smaller squares represent 10 m depth intervals from the land surface to 100 m; the larger squares represent 20 m intervals from 100 m to 300 m below the land surface.

The Tulare Basin is located in south-central California.

(i) A hydrogeologic cross section presented in Plate 1 by Croft (1999) suggests that the aquifer system includes a series of low permeability units, including the “E-clay” (primarily the Corcoran clay unit).

(ii) We analysed wells within the study area that the USGS has defined as either unconfined or confined. Most (>80%) wells at depths of 160-180 m and wells with depths exceeding 160 m are defined as tapping a confined aquifer.

(iii) Croft (1999) state (quote) “Although the E clay is the principal confining bed in the valley (Davis and other, 1959, p. 87-90; Davis and Poland, 1957, p. 426), water-level data indicate that the A clay also is an effective confining bed throughout most of its extent.”

Depth to confined conditions: 160-180 m based on (ii) above

Reference: Croft, M.G. (1999). *Subsurface geology of the late Tertiary and Quaternary water-bearing deposits of the southern part of the San Joaquin Valley, California*. U.S. Geological Survey Water-Supply Paper 1999-H, 35 pp. Accessed April 10, 2022 via <https://pubs.usgs.gov/wsp/1999h/report.pdf>

The table below presents a series of published quotes (see quotation marks denoting text quoted from another publication, which is cited following the quotation marks with the full reference written in full below the table). The leftmost column lists a title of a hydrogeologic formation depicted in the cross section on the previous page. The rightmost column presents a quote from a hydrogeological study (see base of table for citation). The quote has been annotated with colored text to highlight how we categorized each layer (i.e., see categories in the center column in the table). Specifically: (i) blue text highlights portions of a quote that provide insights into the degree of consolidation of the formation, (ii) red text highlights portions of a quote that categorize the formation as an aquifer or an aquitard (i.e., higher versus lower permeability in the context of local hydrogeologic formations), and (iii) green text highlights portions of a quote that provide information about the lithology of the formation.

Supplementary Table 6. Hydrostratigraphy details for the Tulare Basin

Formation name	Category	Quote
Flood-basin, lacustrine, marsh deposits and clay layer	Sedimentary rock aquitard (consolidated or semi-consolidated rock)	"They consist of intercalated lenses of bluish-gray, brown, and reddish-brown fine sand, silt, and clay. In the subsurface, the deposits are interbedded with the younger alluvium and probably in part with the older alluvium. " (Page and Balding, 1973). Figure 5 in Page and Balding, 1973, the explanation shows that " Flood-basin deposits " under " Unconsolidated rocks ". "As indicated on drillers' logs, they range in thickness from 0 to about 100 feet. Because of their impermeable clayey nature , the flood-basin deposits would yield only very small quantities of water to wells " (Page and Balding, 1973).
Alluvium	Unconsolidated aquifer	"The reduced deposits of the alluvium consist of moderately permeable bluish-green fine to coarse sand, silt, and clay. " (Croft, 1999).
Continental and alluvial deposits	Unconsolidated aquifer	" Continental and alluvial deposits of Tertiary and Quaternary age that were derived from the Coast Ranges consist mainly of poorly to moderately permeable yellowish-brown gravel, sand, silt, and clay. " (Croft, 1999).
Continental deposits	Unconsolidated aquifer	"They consist of a gently southwestward-dipping alluvium of poorly sorted gravel, sand, silt, and clay. And they are generally finer grained than the overlying older alluvium (fig. 12)." (Page and Balding, 1973). Figure 5 in Page and Balding, 1973, the explanation shows that " Continental deposits " under " Unconsolidated rocks ". " Yields to wells are as large as 281 cfm (2,100 gpm) and specific capacities as large as 3 ft²/min (22 gpm per foot). " (Page and Badling, 1973).
Basement complex	Endogenous bedrock	"The basement complex forms much of the southern Sierra Nevada, Tehachapi, and San Emigdio Mountains and is composed of a mass of plutonic and metamorphic rocks commonly referred to as the Sierra Nevada batholith of pre-Tertiary age. " Croft, M.G. (1999). " The rocks of the basement complex are of little importance as a source of ground water because they are largely impermeable and

Formatted: Font: Arial, 10 pt,

Formation name	Category	Quote
		generally lie beneath most water wells, except locally along the eastern margin of the valley. (Croft, 1999).

Page, R. W., Balding, G. O. (1973). Geology and quality of water in the Modesto-Merced area, San Joaquin Valley, California, with a brief section on hydrology. (No.6-73). US Geological Survey Water-Resources Investigations Report 73-6, 77 pp. <https://pubs.er.usgs.gov/publication/wri736>

Croft, M.G. (1999). Subsurface geology of the late Tertiary and Quaternary water-bearing deposits of the southern part of the San Joaquin Valley, California. U.S. Geological Survey Water-Supply Paper 1999-H, 35 pp. Accessed June 17, 2022 via <https://pubs.usgs.gov/wsp/1999h/report.pdf>

3.4 Eastern Carrizo-Wilcox, Carrizo-Wilcox

Supplementary Fig. 105. Hydrogeologic cross section. 20 equally spaced transparent pink bars overlie the cross section; each shaded bar depicts the vertical offset from the land surface to the top of the uppermost confining unit or endogenous bedrock.

Supplementary Fig. 106. Vertical variations in the prevalence of wells that have been defined as tapping an unconfined or a confined aquifer by the USGS. The smaller squares represent 10 m depth intervals from the land surface to 100 m; the larger squares represent 20 m intervals from 100 m to 300 m below the land surface.

The Eastern Carrizo-Wilcox aquifer system is located in eastern Texas and northwestern Louisiana.

(i) A hydrogeologic cross section presented in Fig. 2.4 by George (2009) depicts a series of confining units including the relatively shallow Reklaw Formation and the Weches Formation.

(ii) We analysed wells within the study area that the USGS has defined as either unconfined or confined. Most (>80%) wells at depths of 20-30 m and at depths exceeding 20 m are defined as tapping a confined aquifer.

Depth to confined conditions: 20-30 m based on (ii) above

Reference: George, P.G. (2009). Geology of the Carrizo-Wilcox Aquifer. In: *Aquifers of the upper coastal plains of Texas* (eds. Hutchison, W.R., Davidson, S.C., Brown, B.J., Mace, R.E.), Chapter 2, Texas Water Development Board Report 374, pp. 17-34. Accessed May 19, 2021 from https://www.twdb.texas.gov/publications/reports/numbered_reports/doc/R374_AquifersofUpperCoastalPlains.pdf

The table below presents a series of published quotes (see quotation marks denoting text quoted from another publication, which is cited following the quotation marks with the full reference written in full below the table). The leftmost column lists a title of a hydrogeologic formation depicted in the cross section on the previous page. The rightmost column presents a quote from a hydrogeological study (see base of table for citation). The quote has been annotated with colored text to highlight how we categorized each layer (i.e., see categories in the center column in the table). Specifically: (i) blue text highlights portions of a quote that provide insights into the degree of consolidation of the formation, (ii) red text highlights portions of a quote that categorize the formation as an aquifer or an aquitard (i.e., higher versus lower permeability in the context of local hydrogeologic formations), and (iii) green text highlights portions of a quote that provide information about the lithology of the formation.

Supplementary Table 7. Hydrostratigraphy details for the Eastern Carrizo-Wilcox

Formation name	Category	Quote
Younger Formation	Unconsolidated aquifer	" Sand, silt, clay, and some gravel. " (Sandeen 1987) (Table 6). " May yield small quantities of water to shallow dug wells. " (Sandeen 1987). " Sand, silt, and clay " (Sandeen 1987)
Sparta Sand	Sedimentary rock aquifer (consolidated or semi-consolidated rock)	" Interbedded sand, clay, and silt. " (Sandeen 1987) (Table 6). " Feeds springs; may yield some water to dug wells. " (Sandeen 1987).
Weches Formation	Sedimentary rock aquitard (consolidated or semi-consolidated rock)	" Glauconite, glauconitic clay and sand. Secondary deposits of limestone in outcrop. " (Sandeen 1987) (Table 6). " Not known to yield water to wells in Rusk County. (Sandeen 1987)
Queen City Sand	Sedimentary rock aquifer (consolidated or semi-consolidated rock)	" Sand, silt, clay, and some lignite. " (Sandeen 1987) (Table 6). " Yields small to moderate quantities of freshwater. " (Sandeen 1987) "The Queen City Formation is composed of sand, sandstone, shale, and clay, with lignite found locally. The aquifer thickness is less than 500 feet in most places but reaches almost 700 feet in parts of northeast Texas. " (Davidson et al. 2009)
Reklaw Formation	Sedimentary rock aquitard (consolidated or semi-consolidated rock)	" Glauconitic clay, some sand, weathers to a red clayey soil, limonite seams, iron concretions. " (Sandeen 1987) (Table 6). " Yields small quantities of water to wells. " (Sandeen 1987) "The Carrizo-Wilcox aquifer is separated from the overlying Queen City aquifer by the Reklaw or Bigford Fm., which is a confining unit (Fig. 2)." (Huang et al. 2012). "It is separated from the overlying Queen City Aquifer by the relatively impermeable Reklaw Formation." (George, 2009).
Carrizo Sand	Sedimentary rock aquifer (consolidated or semi-consolidated rock)	" Grey to white. Often massive sand, clay lenses; may be predominantly clayey. " (Sandeen 1987) (Table 6). " Yields large to moderate quantities of freshwater. In hydrologic continuity with the Wilcox. " (Sandeen 1987)
Wilcox Group	Sedimentary rock aquifer (consolidated or semi-consolidated rock)	"Thin, sometimes massive beds of sand; clay and lignite. Beds often dis-continuous." (Sandeen 1987) (Table 6). " Yields large to moderate quantities of freshwater. In hydrologic continuity with the Wilcox. " (Sandeen 1987)

Formation name	Category	Quote
Midway	Sedimentary rock aquitard (consolidated or semi-consolidated rock)	"Calcareous clay and minor amounts of limestone, silt, and glauconitic clay. " (Sandeen 1987) (Table 6). "Not known to yield water to wells in Rusk County; upper sand may contain some slightly saline water." (Sandeen 1987)

Sandeen, W. M. (1984). Ground-water resources of Rusk County, Texas. US Geological Survey. (<https://pubs.usgs.gov/of/1983/0757/report.pdf>)

Huang, Y., Scanlon, B. R., Nicot, J. P., Reedy, R. C., Dutton, A. R., Kelley, V. A., Deeds, N. E. (2012). Sources of groundwater pumpage in a layered aquifer system in the Upper Gulf Coastal Plain, USA. *Hydrogeology Journal*, 20(4), 783-796.

George, P.G. (2009). Geology of the Carrizo-Wilcox Aquifer. In: *Aquifers of the upper coastal plains of Texas* (eds. Hutchison, W.R., Davidson, S.C., Brown, B.J., Mace, R.E.), Chapter 2, Texas Water Development Board Report 374, pp. 17-34. Accessed June 14, 2022 from https://www.twdb.texas.gov/publications/reports/numbered_reports/doc/R374_AquifersofUpperCoastalPlains.pdf

Sarah C. Davidson, Brenner J. Brown, and Robert E. Mace (2009). Geology of the Carrizo-Wilcox Aquifer. In: *Aquifers of the upper coastal plains of Texas* (eds. Hutchison, W.R., Davidson, S.C., Brown, B.J., Mace, R.E.), Chapter 1, Texas Water Development Board Report 374, pp. 1-16. Accessed June 14, 2022 from https://www.twdb.texas.gov/publications/reports/numbered_reports/doc/R374_AquifersofUpperCoastalPlains.pdf

3.5 Eagle Valley, Carson River Basin

Supplementary Fig. 107. Hydrogeologic cross section. 20 equally spaced transparent pink bars overlie the cross section; each shaded bar depicts the vertical offset from the land surface to the top of the uppermost confining unit or endogenous bedrock.

Supplementary Fig. 108. Vertical variations in the prevalence of wells that have been defined as tapping an unconfined or a confined aquifer by the USGS. The smaller squares represent 10 m depth intervals from the land surface to 100 m; the larger squares represent 20 m intervals from 100 m to 300 m below the land surface.

Eagle Valley is located in western Nevada.

(i) To create a hydrogeologic cross section, we georeferenced Fig. 11 (pp. 25) by Arteaga (1986) that displays the (quote) “(a)pproximate thickness of valley-fill sedimentary deposits”.

(ii) Only half of the deepest wells that the USGS has defined as confined/unconfined (depths exceeding 200 m) are defined as

tapping a confined aquifer: the deepest well in the dataset (381 m) is defined as tapping an unconfined aquifer.

(iii) Thiros et al. (2010; see section 4 by J.M. Huntington) state (quote): “Unconfined to confined conditions are present in the basin-fill sediments... The degree of confinement varies spatially through the valley due to the clay lenses being discontinuous at different depths.” and “Although groundwater exists under both confined and unconfined conditions in Carson Valley, no single confining layer extends across the entire valley... the confining layers occur mainly as scattered, discontinuous clay beds, 30 to 70 ft thick, at a depth of 200 to 300 ft.”

Depth to confined conditions: >200 m based on (ii)

References: Arteaga, F.E. (1986). Mathematical model analysis of the Eagle Valley Ground-water basin, west-central Nevada, US Geological Survey Water-Resources Bulletin 45, 65 pp. Accessed May 20, 2022 via <http://images.water.nv.gov/images/publications/water%20resources%20bulletins/Bulletin45.pdf>

Thiros, S.A., Bexfield, L.M., Anning, D.W., and Huntington, J.M., eds., (2010). Conceptual understanding and groundwater quality of selected basin-fill aquifers in the Southwestern United States: US Geological Professional Paper 1781, 288 pp. Accessed May 19, 2022 via <https://pubs.usgs.gov/pp/1781/>

The table below presents a series of published quotes (see quotation marks denoting text quoted from another publication, which is cited following the quotation marks with the full reference written in full below the table). The leftmost column lists a title of a hydrogeologic formation depicted in the cross section on the previous page. The rightmost column presents a quote from a hydrogeological study (see base of table for citation). The quote has been annotated with colored text to highlight how we categorized each layer (i.e., see categories in the center column in the table). Specifically: (i) blue text highlights portions of a quote that provide insights into the degree of consolidation of the formation, (ii) red text highlights portions of a quote that categorize the formation as an aquifer or an aquitard (i.e., higher versus lower permeability in the context of local hydrogeologic formations), and (iii) green text highlights portions of a quote that provide information about the lithology of the formation.

Supplementary Table 8. Hydrostratigraphy details for the Eagle Valley

Formation name	Category	Quote
Basin-fill sediments	Unconsolidated aquifer	"Quaternary sediments of two ages are present in Eagle Valley. The older sediments form fans at the mouths of deeply incised canyons on the western side of the valley. Small individual fans merge into one wide fan extending as much as 1 mi eastward into the valley from the mountain front and are made up of partly consolidated to unconsolidated gravel, sand, and silt, with discontinuous clay layers (Maurer and others, 1996). Similar fans are present at the base of the Virginia Range to the north and Prison Hill to the east (Trexler and others, 1980). The discontinuity of clay layers in the central part of the basin enable a direct hydraulic connection from the land surface to the basin-fill aquifer and make the aquifer susceptible to contamination from sources at the surface (Lico, 1998, p. 1). The younger sediments in the valley lowlands consist of fine-grained sands, silty and muddy sands, and clay (Arteaga, 1986; Trexler and others, 1980). Overall, basin-fill sediments are coarse-grained near the base of the mountains and finer grained near the center of the valley. The basin-fill sediments are estimated to be about 1,200 ft thick at a point 1.5 mi west of Lone Mountain, about 400 to 800 ft thick beneath the northeastern and southern parts of Eagle Valley, and about 2,000 ft thick about 1 mi northwest of Prison Hill (Arteaga, 1986). In general, the deepest part of the alluvial basin is in the center of the Eagle Valley (Schaefer and others, 2007)." quoting Thiros et al. (2010)
Volcanic rocks	Volcanic rocks (see Fig. 1 by Thiros et al. (2010) on page 50)	"Mesozoic-age granite and metamorphosed rocks crop out to the north and west of Eagle Valley and near Prison Hill, and most likely underlie most of the valley floor (Moore, 1969). In the Virginia Range, Tertiary sandstone and volcanic rocks consisting mostly of rhyolite, andesite, and basalt flows, flow breccias, and tuffs overlie the granite and metamorphosed rocks (Moore, 1969; Trexler, 1977)." quoting Thiros et al. (2010)
Undifferentiated sedimentary rocks	Sedimentary aquitard	Based on Fig. 1 by Thiros et al. (2010)

3.6 Central Wabash and Bloomington Ridged Plain, Central Lowland Till Plain

Supplementary Fig. 109. Hydrogeologic cross section. 20 equally spaced transparent pink bars overlie the cross section; each shaded bar depicts the vertical offset from the land surface to the top of the uppermost confining unit or endogenous bedrock.

Supplementary Fig. 110. Vertical variations in the prevalence of wells that have been defined as tapping an unconfined or a confined aquifer by the USGS. The smaller squares represent 10 m depth intervals from the land surface to 100 m; the larger squares represent 20 m intervals from 100 m to 300 m below the land surface.

The Central Wabash and Bloomington Ridged Plain is located in eastern Illinois.

(i) A hydrogeologic cross section presented in Fig. 2 by Panno et al. (1994) depicts sedimentary aquifers at shallow depths (including the Mahomet Sand member of the Banner Formation) overlying low permeability layers. In some areas carbonate rocks directly underlie the shallow sedimentary aquifers.

(ii) We analysed wells within the study area that the USGS has defined as either unconfined or confined. Most (>80%) wells at depths of 30-40 m and at depths exceeding 30 m are defined as tapping a confined aquifer.

Depth to confined conditions: 30-40 m (based on (ii) above)

Reference: Panno, S. V., Hackley, K. C., Cartwright, K., Liu, C. L. (1994). Hydrochemistry of the Mahomet Bedrock Valley Aquifer, east-central Illinois: Indicators of recharge and ground-water flow. *Groundwater*, 32, 591-604.

The table below presents a series of published quotes (see quotation marks denoting text quoted from another publication, which is cited following the quotation marks with the full reference written in full below the table). The leftmost column lists a title of a hydrogeologic formation depicted in the cross section on the previous page. The rightmost column presents a quote from a hydrogeological study (see base of table for citation). The quote has been annotated with colored text to highlight how we categorized each layer (i.e., see categories in the center column in the table). Specifically: (i) blue text highlights portions of a quote that provide insights into the degree of consolidation of the formation, (ii) red text highlights portions of a quote that categorize the formation as an aquifer or an aquitard (i.e., higher versus lower permeability in the context of local hydrogeologic formations), and (iii) green text highlights portions of a quote that provide information about the lithology of the formation.

Supplementary Table 9. Hydrostratigraphy details for the Central Wabash Bloomington Ridge Plain

Formation name	Category	Quote
Wedron Formation	Unconsolidated aquifer	"Water-bearing sand and gravel deposits contained within the Wedron Formation occur only as scattered pockets, as the formation consists principally of glacial till." Anliker & Sanderson, 1995). "The chemistry of the ground waters from the sands of the Wedron, Glasford, and Banner Formations is very similar, suggesting that the aquifers of these formations are in hydraulic communication (Figure 2)." (Panno et al., 1994).
Glasford Formation	Unconsolidated aquifer	"The MVA is generally coincident with the Mahomet Sand Member of the Banner Formation, and is overlain by the Glasford and the Wedron Formations (Figure 2). Sand and gravel outwash deposits within the overlying Glasford Formation are usually most extensive where they are associated with the Vandalia Till Member and constitute a second productive aquifer that is locally important." (Panno et al., 1994).
Banner Formation	Sedimentary rock aquifer (consolidated or semi-consolidated rock)	"The chemistry of the ground waters from the sands of the Wedron, Glasford, and Banner Formations is very similar, suggesting that the aquifers of these formations are in hydraulic communication (Figure 2)." (Panno et al., 1994). "The Mahomet Sand Member is composed of glacial outwash sand and gravel laid down within the confines of the Mahomet Valley Lowland." (Panno et al., 1994).
Pennsylvanian	Sedimentary rock aquitard (consolidated or semi-consolidated rock)	"The bedrock geology of the study area consists of Pennsylvanian, Mississippian, Devonian, and Silurian rocks. The western half of the valley is dominated by Pennsylvanian rocks (typically shale with thin limestone, sandstone, and coal). The eastern part of the valley is underlain by rocks of the Pennsylvanian, Mississippian, and Devonian." (Panno et al., 1994).
Coal	Sedimentary rock aquitard (consolidated or semi-consolidated rock)	"The bedrock geology of the study area consists of Pennsylvanian, Mississippian, Devonian, and Silurian rocks. The western half of the valley is dominated by Pennsylvanian rocks (typically shale with thin limestone, sandstone, and coal). The eastern part of the valley is underlain by rocks of the Pennsylvanian, Mississippian, and Devonian." (Panno et al., 1994). "(Figure 2) consists of a subcrop of Pennsylvanian coals and associated black shale. Coals in this area are thin and typically contain several percent S" (Panno et al., 1994).

Formation name	Category	Quote
Mississippian	Carbonate aquifer	"The bedrock geology of the study area consists of Pennsylvanian, Mississippian , Devonian, and Silurian rocks. The western half of the valley is dominated by Pennsylvanian rocks (typically shale with thin limestone, sandstone, and coal). The eastern part of the valley is underlain by rocks of the Pennsylvanian, Mississippian, and Devonian." (Panno et al., 1994). " Shale, siltstone, limestone and dolomite; upper part is soft, fissile, and fractured; locally contains siderite nodules, plant macrofossils, and slickensides " (Stumpf, 2018).
Devonian	Carbonate aquifer	"The bedrock geology of the study area consists of Pennsylvanian, Mississippian, Devonian , and Silurian rocks. The western half of the valley is dominated by Pennsylvanian rocks (typically shale with thin limestone, sandstone, and coal). The eastern part of the valley is underlain by rocks of the Pennsylvanian, Mississippian, and Devonian. " (Panno et al., 1994). "The Onarga Valley, a northeastern tributary of the MVA , is dominated by bedrock of Middle Devonian and Silurian age strata that consists of dolomite and the Pennsylvanian Caseyville, Abbott, and Spoon Formations that contain thin coals, black shale, and thin limestone (Willman et al., 1967)." (Panno et al., 1994). " Mahomet Bedrock Valley Aquifer (MVA) " (Panno et al., 1994).
Silurian	Carbonate aquifer	"The bedrock geology of the study area consists of Pennsylvanian, Mississippian, Devonian, and Silurian rocks. The western half of the valley is dominated by Pennsylvanian rocks (typically shale with thin limestone, sandstone, and coal). The eastern part of the valley is underlain by rocks of the Pennsylvanian, Mississippian, and Devonian." (Panno et al., 1994). " Ground water from most wells in the Silurian dolomite in Iroquois Co. contain relatively low concentrations of SO ₄ ²⁻ ." (Panno et al., 1994).

Leighton, M. M., Ekblaw, G. E., Horberg, L. (1948). Physiographic divisions of Illinois. The Journal of Geology, 56, 16-33.

Anliker, M. A., Sanderson, E. W. (1995). Reconnaissance study of ground-water levels and withdrawals in the vicinity of DeWitt and Piatt counties. ISWS Contract Report CR 589. Accessed July 10, 2022 via <http://citeseerx.ist.psu.edu/viewdoc/download?doi=10.1.1.542.7107&rep=rep1&type=pdf>

Stumpf, A.J. (2018). Geologic cross sections of Quaternary deposits across the Manlove gas storage field area, Champaign County, Illinois: Illinois State Geological Survey, Special Report 6, 7 p.; 2 plates. Accessed July 10, 2022 via <https://www2.illinois.gov/epa/Documents/iepa/community-relations/mahomet-aquifer/task-force/Special-Report-6.pdf>

3.7 Palouse Slope, Columbia Plateau Regional Aquifer System

Supplementary Fig. 111. Hydrogeologic cross section. 20 equally spaced transparent pink bars overlie the cross section; each shaded bar depicts the vertical offset from the land surface to the top of the uppermost confining unit or endogenous bedrock.

Supplementary Fig. 112. Vertical variations in the prevalence of wells that have been defined as tapping an unconfined or a confined aquifer by the USGS. The smaller squares represent 10 m depth intervals from the land surface to 100 m; the larger squares represent 20 m intervals from 100 m to 300 m below the land surface.

The Palouse Slope is located in central Washington State.

(i) We examined a hydrogeologic cross section generated using a web application available here: <https://or.water.usgs.gov/proj/cpras/index.html>. No clear confining unit is depicted on the cross section.

(ii) We analysed wells within the study area that the USGS has defined as either unconfined or confined. Only half of the deepest wells in our dataset (depths exceeding 200 m) are defined as tapping a confined aquifer; the deepest well in the dataset (276 m) is defined as tapping an unconfined aquifer.

Depth to confined conditions: >276 m based on (ii) (consistent with hydrogeologic cross section depicted in figure to the left)

The table below presents a series of published quotes (see quotation marks denoting text quoted from another publication, which is cited following the quotation marks with the full reference written in full below the table). The leftmost column lists a title of a hydrogeologic formation depicted in the cross section on the previous page. The rightmost column presents a quote from a hydrogeological study (see base of table for citation). The quote has been annotated with colored text to highlight how we categorized each layer (i.e., see categories in the center column in the table). Specifically: (i) blue text highlights portions of a quote that provide insights into the degree of consolidation of the formation, (ii) red text highlights portions of a quote that categorize the formation as an aquifer or an aquitard (i.e., higher versus lower permeability in the context of local hydrogeologic formations), and (iii) green text highlights portions of a quote that provide information about the lithology of the formation.

Supplementary Table 10. Hydrostratigraphy details for Palouse Slope

Formation name	Category	Quote
Saddle Mountains Basalts	Volcanic aquifer	"The Saddle Mountains unit consists mostly of the Saddle Mountains Basalt and interbed members. " (Burns et al., 2012). " The Saddle Mountains unit Kh has been estimated to range from 0.007 to 3,200 ft/d, with a median of about 1 to 2 ft/d. " (Burns et al., 2012).
Wanapum Basalt	Volcanic aquifer	"The Saddle Mountains unit consists mostly of the Saddle Mountains Basalt and interbed members. " (Burns et al., 2012). " The Saddle Mountains unit Kh has been estimated to range from 0.007 to 3,200 ft/d, with a median of about 1 to 2 ft/d. " (Burns et al., 2012).
Grande Ronde Basalt	Volcanic aquifer	"The Grande Ronde unit is the oldest and most extensive of the basalt units. " (Burns et al., 2012). " The range in Grande Ronde unit Kh values was similar to that for the Wanapum unit, from 0.005 to 5,200 ft/d. Median Kh values for the Grande Ronde unit range from about 0.1 to 5 ft/d (again, excluding value for flow interior). " (Burns et al., 2012).

Burns, E. R., Snyder, D. T., Haynes, J. V., Waibel, M. S. (2012). Groundwater status and trends for the Columbia Plateau Regional Aquifer System, Washington, Oregon, and Idaho. U.S. Geological Survey Scientific Investigations Report 2012–5261, 52 pp. Accessed February 18, 2021 from <http://pubs.er.usgs.gov/publication/sir20125261>

3.8 Umatilla Basin and Horse Heaven Hills, Columbia Plateau Regional Aquifer System

Supplementary Fig. 113. Hydrogeologic cross section. 20 equally spaced transparent pink bars overlie the cross section; each shaded bar depicts the vertical offset from the land surface to the top of the uppermost confining unit or endogenous bedrock.

Supplementary Fig. 114. Vertical variations in the prevalence of wells that have been defined as tapping an unconfined or a confined aquifer by the USGS. The smaller squares represent 10 m depth intervals from the land surface to 100 m; the larger squares represent 20 m intervals from 100 m to 300 m below the land surface.

The Umatilla Basin and Horse Heaven Hills are located in southern Washington and northern Oregon.

(i) A hydrogeologic cross section presented in Fig. 3 by Herrera et al. (2017) suggests that the system does not have a clear confining unit. However, the system is confined by basalt flow interiors (see (iii) below).

(ii) We analysed wells within the study area that the USGS has defined as either unconfined or confined. All (100%) wells at depths of 60-70 m and at depths exceeding 60 m are defined as tapping a confined aquifer. No wells in our dataset have depths between 25 m and 69 m; all wells (n=2) with depths of shallower than 25 m are classified as unconfined.

(iii) Herrera et al. (2017) state that (quote): "The uppermost part of the CRBG is often permeable and unconfined, and has a good hydraulic connection with the overlying alluvial aquifer and, in some cases, streams. Permeable interflow zones at depth are confined by the flow interiors."

Depth to confined conditions: 60-70 m (based on (ii) above)

Reference: Herrera, N. B., Ely, K., Mehta, S., Stonewall, A. J., Risley, J. C., Hinkle, S. R., Conlon, T. D. (2017). Hydrogeologic framework and selected components of the groundwater budget for the upper Umatilla River Basin, Oregon. US Geological Survey Scientific Investigations Report 2017-5020, 68 pp. Accessed February 18, 2021 from <https://pubs.usgs.gov/sir/2017/5020/sir20175020.pdf>

The table below presents a series of published quotes (see quotation marks denoting text quoted from another publication, which is cited following the quotation marks with the full reference written in full below the table). The leftmost column lists a title of a hydrogeologic formation depicted in the cross section on the previous page. The rightmost column presents a quote from a hydrogeological study (see base of table for citation). The quote has been annotated with colored text to highlight how we categorized each layer (i.e., see categories in the center column in the table). Specifically: (i) blue text highlights portions of a quote that provide insights into the degree of consolidation of the formation, (ii) red text highlights portions of a quote that categorize the formation as an aquifer or an aquitard (i.e., higher versus lower permeability in the context of local hydrogeologic formations), and (iii) green text highlights portions of a quote that provide information about the lithology of the formation.

Supplementary Table 11. Hydrostratigraphy details for Umatilla Basin and Horse Heaven Hills

Formation name	Category	Quote
Sediment	Unconsolidated aquifer (large proportion of silt results in low permeability)	"The sedimentary deposits overlying the CRBG consist of four types of Tertiary and Quaternary sediments defined by Hogenson (1964) and refined by Ferns and Ely (2006) (figs. 3 and 6): (1) recent stream alluvium, including Pleistocene terrace deposits (Qal), (2) Quaternary alluvial fan deposits (Qf), (3) Quaternary landslide deposits (Qls), (4) loess and fine-grained sandstone (QTs) interpreted as late Miocene wind-reworked, fine grained deposits correlative to the McKay Formation, and (5) late-Miocene to early-Pliocene conglomerate of the McKay Formation (Tms) (fig. 3). " (Herrera et al., 2017). " Mean hydraulic conductivity and storage coefficient values for the sedimentary unit were determined from well data in the Hermiston-Umatilla area and estimated to be 24,000 ft/d and 0.15, respectively (Davies-Smith and others, 1988)." (Herrera et al., 2017).
Saddle Mountains	Volcanic aquifer (upper most part is unconfined)	" Conceptually, the CRBG is a series of productive aquifers consisting of relatively high permeability interflow zones separated by the low permeability flow interiors. The uppermost part of the CRBG is often permeable and unconfined, and has a good hydraulic connection with the overlying alluvial aquifer and, in some cases, streams. " (Herrera et al., 2017). "Mean values of hydraulic conductivity determined from about 1,700 short-duration specific capacity tests in the Umatilla River Basin are 18 ft/d for the Saddle Mountains basalt unit, 170 ft/d for the Wanapum basalt unit, and 65 ft/d for the Grande Ronde basalt unit (Davies-Smith and others, 1988)." (Herrera et al., 2017).
Wanapum basalt unit	Volcanic aquifer	" Conceptually, the CRBG is a series of productive aquifers consisting of relatively high permeability interflow zones separated by the low permeability flow interiors. The uppermost part of the CRBG is often permeable and unconfined, and has a good hydraulic connection with the overlying alluvial aquifer and, in some cases, streams. " (Herrera et al., 2017). "Mean values of hydraulic conductivity determined from about 1,700 short-duration specific capacity tests in the Umatilla River Basin are 18 ft/d for the Saddle Mountains basalt unit, 170 ft/d for the Wanapum basalt unit, and 65 ft/d for the Grande Ronde basalt unit (Davies-Smith and others, 1988)." (Herrera et al., 2017).

Formation name	Category	Quote
Grande Ronde basalt unit	Volcanic aquifer	"Conceptually, the CRBG is a series of productive aquifers consisting of relatively high permeability interflow zones separated by the low permeability flow interiors. The uppermost part of the CRBG is often permeable and unconfined, and has a good hydraulic connection with the overlying alluvial aquifer and, in some cases, streams." (Herrera et al., 2017). "Mean values of hydraulic conductivity determined from about 1,700 short-duration specific capacity tests in the Umatilla River Basin are 18 ft/d for the Saddle Mountains basalt unit, 170 ft/d for the Wanapum basalt unit, and 65 ft/d for the Grande Ronde basalt unit (Davies-Smith and others, 1988)." (Herrera et al., 2017).

Herrera, N.B., Ely, K. Mehta, S., Stonewall, A.J., Risley, J.C., Hinkle, S.R., Conlon, T.D. (2017). Hydrogeologic framework and selected components of the groundwater budget for the upper Umatilla River Basin, Oregon. No. 2017-5020. US Geological Survey. <https://pubs.er.usgs.gov/publication/sir20175020>

3.9 Yakima Basin, Columbia Plateau Regional Aquifer System

Supplementary Fig. 115. Hydrogeologic cross section. 20 equally spaced transparent pink bars overlie the cross section; each shaded bar depicts the vertical offset from the land surface to the top of the uppermost confining unit or endogenous bedrock.

Supplementary Fig. 116. Vertical variations in the prevalence of wells that have been defined as tapping an unconfined or a confined aquifer by the USGS. The smaller squares represent 10 m depth intervals from the land surface to 100 m; the larger squares represent 20 m intervals from 100 m to 300 m below the land surface.

The Yakima Basin is located in central Washington State.

(i) A hydrogeologic section by Kahle et al. (2011) in their Fig. 7b demonstrates the high degree of topographic and geologic complexity in the Yakima Basin. Their cross section suggests that the uppermost confining unit is 282 meters below land surface (25th-75th percentile range: 209 m to >529 m meters below land surface)

(ii) We analysed wells within the study area that the USGS has defined as either unconfined or confined. Three or the four deepest wells in our dataset (depths exceeding 200 m) are defined as tapping a confined aquifer; the deepest well in the dataset (291 m, located on the northeast periphery of Grandview) is defined as tapping an unconfined aquifer.

Depth to confined conditions: 280-300 m (based on (i) above)

Reference: Kahle, S.C., Morgan, D.S., Welch, W.B., Ely, S.R., Vaccaro, J.J., Orzol, L.L. (2011). Hydrogeologic Framework and Hydrologic Budget Components of the Columbia Plateau Regional Aquifer System, Washington, Oregon, and Idaho. US Geological Survey Scientific Investigations Report 2011-5124, 80 pp. Accessed February 16, 2021 from <https://pubs.usgs.gov/sir/2011/5124/pdf/sir20115123.pdf>

The table below presents a series of published quotes (see quotation marks denoting text quoted from another publication, which is cited following the quotation marks with the full reference written in full below the table). The leftmost column lists a title of a hydrogeologic formation depicted in the cross section on the previous page. The rightmost column presents a quote from a hydrogeological study (see base of table for citation). The quote has been annotated with colored text to highlight how we categorized each layer (i.e., see categories in the center column in the table). Specifically: (i) blue text highlights portions of a quote that provide insights into the degree of consolidation of the formation, (ii) red text highlights portions of a quote that categorize the formation as an aquifer or an aquitard (i.e., higher versus lower permeability in the context of local hydrogeologic formations), and (iii) green text highlights portions of a quote that provide information about the lithology of the formation.

Supplementary Table 12. Hydrostratigraphy details for the Yakima Basin

Formation name	Category	Quote
Overburden	Unconsolidated aquifer (consolidated aquifer also exists)	"Overburden deposits are diverse in lithology and, thus, so are their hydraulic characteristics (table 2). The deposits, which consist of unconsolidated and consolidated material of alluvial, glacial, lacustrine, wind-blown, and volcanic origins, form important water-bearing units, as well as semiconfining to confining units. " (Burns et al., 2012).
Saddle Mountains	Volcanic aquifer	"The Saddle Mountains unit consists mostly of the Saddle Mountains Basalt and interbed members. " (Burns et al., 2012). " The Saddle Mountains unit Kh has been estimated to range from 0.007 to 3,200 ft/d, with a median of about 1 to 2 ft/d. " (Burns et al., 2012).
Mabton Interbed Unit	Sedimentary rock aquitard (consolidated or semi-consolidated rock) (laterally extensive but thin)	"The Mabton unit generally consists of clay, shale, claystone, clay with basalt, clay with sand, and sandstone. " (Burns et al., 2012). " The confining units are equivalent to the Saddle Mountains-Wanapum and Wanapum-Grande Ronde interbeds, referred to in this study as the Mabton and Vantage Interbeds, respectively." (Burns et al., 2012).
Wanapum	Volcanic aquifer	"The Wanapum unit, composed mostly of basalt and interbed members of the Wanapum basalt, is in most of the north-central part of the study area" (Burns et al., 2012). " The Wanapum unit had a slightly larger range (0.007 to 5,200 ft/d) than the Saddle Mountains unit, and the median reported Kh for the Wanapum unit ranges from about 3 to 11 ft/d (excluding value for flow interior). " (Burns et al., 2012).
Vantage Interbed	Sedimentary rock aquitard (consolidated or semi-consolidated rock) (laterally extensive but thin)	"The Vantage unit is the sedimentary interbed between the overlying Wanapum unit and the underlying Grande Ronde unit." (Burns et al., 2012). " The confining units are equivalent to the Saddle Mountains-Wanapum and Wanapum-Grande Ronde interbeds, referred to in this study as the Mabton and Vantage Interbeds, respectively." (Burns et al., 2012).
Grande Ronde	Volcanic aquifer	"The Grande Ronde unit is the oldest and most extensive of the basalt units. " (Burns et al., 2012). " The range in Grande Ronde unit Kh values was similar to that for the Wanapum unit, from 0.005 to 5,200 ft/d. Median Kh values for the Grande Ronde unit range from about 0.1 to 5 ft/d (again, excluding value for flow interior). " (Burns et al., 2012).

Formation name	Category	Quote
Older bedrock	Endogenous bedrock (water may present in the bedrock)	"The areas bordering and underlying the CPRAS include metamorphic (crystalline), sedimentary, volcanic, and intrusive and extrusive igneous rocks . In general, the older bedrock has lower values of porosity and permeability than the overburden and CRBG units." (Burns et al., 2012).

Burns, E. R., Snyder, D. T., Haynes, J. V., Waibel, M. S. (2012). Groundwater status and trends for the Columbia Plateau Regional Aquifer System, Washington, Oregon, and Idaho. U.S. Geological Survey Scientific Investigations Report 2012-5261, 52 pp. Accessed February 18, 2021 from <http://pubs.er.usgs.gov/publication/sir20125261>

3.10 Stockton Plateau, Edwards-Trinity Aquifer System

Supplementary Fig. 117. Hydrogeologic cross section. 20 equally spaced transparent pink bars overlie the cross section; each shaded bar depicts the vertical offset from the land surface to the top of the uppermost confining unit or endogenous bedrock.

Supplementary Fig. 118. Vertical variations in the prevalence of wells that have been defined as tapping an unconfined or a confined aquifer by the USGS. The smaller squares represent 10 m depth intervals from the land surface to 100 m; the larger squares represent 20 m intervals from 100 m to 300 m below the land surface.

The Stockton Plateau is located in western Texas and forms part of the broader Edwards-Trinity Aquifer System.

(i) A hydrogeologic cross section presented in Fig. 6-9 by Meyer et al. (2012) does not depict a clear confining unit within the aquifer system within the uppermost 190-340 m of the aquifer system (median of pink transparent pink bars suggests a lack of a clear confining unit in the uppermost 249 m below the land surface).

(ii) We analysed wells within the study area that the USGS has defined as either unconfined or confined. Only four wells were available in our dataset; they have depths ranging from 30 m to 87 m, and all are classified as tapping unconfined aquifers.

Depth to confined conditions: >249 meters below land surface (based on (i) above)

Reference: Meyer, J.E., Wise, M.R., Kalaswad, S. (2012). Pecos Valley Aquifer, West Texas: Structure and brackish groundwater. Texas Water Development Board Report 382, 95 pp. Accessed May 15, 2022 via https://www.twdb.texas.gov/publications/reports/numbered_reports/doc/R382_PecosValley.pdf

The table below presents a series of published quotes (see quotation marks denoting text quoted from another publication, which is cited following the quotation marks with the full reference written in full below the table). The leftmost column lists a title of a hydrogeologic formation depicted in the cross section on the previous page. The rightmost column presents a quote from a hydrogeological study (see base of table for citation). The quote has been annotated with colored text to highlight how we categorized each layer (i.e., see categories in the center column in the table). Specifically: (i) blue text highlights portions of a quote that provide insights into the degree of consolidation of the formation, (ii) red text highlights portions of a quote that categorize the formation as an aquifer or an aquitard (i.e., higher versus lower permeability in the context of local hydrogeologic formations), and (iii) green text highlights portions of a quote that provide information about the lithology of the formation.

Supplementary Table 13. Hydrostratigraphy details for Stockton Plateau

Formation name	Category	Quote
Pecos Valley Alluvium	Unconsolidated aquifer (unconfined aquifer, although deeper sections may have local confining layers)	"The Pecos Valley Aquifer , previously known as the Cenozoic Pecos Alluvium , is designated as a major aquifer in Texas (Ashworth and Hopkins, 1995; George and others, 2011)." (Meyer et al., 2012). "The stratigraphic top of the Pecos Valley Alluvium consists of post-Cretaceous sediments that are exposed at ground surface in the study area." (Meyer et al., 2012)
Cretaceous Undivided	Clastic sedimentary rock aquifer (consolidated or semi-consolidated rock) (carbonate rocks can be found)	"The unit mapped as Cretaceous Undivided consists of Cretaceous sediments deposited unconformably on the Triassic Dockum Group or the Permian Dewey Lake Formation (Table 6-1). The sediments consist of clay, sand, and limestone deposited in continental to marine depositional settings at the onset of the marine transgression in West Texas. The undifferentiated Cretaceous Undivided unit constitutes the hydrostratigraphic unit for the Edwards-Trinity (Plateau) Aquifer, classified as a major aquifer in Texas (Ashworth and Hopkins, 1995; George and others, 2011)." (Meyer et al., 2012)
Dockum Group- Dewey Lake Formation	Clastic sedimentary rock aquifer (consolidated or semi-consolidated rock) (water quality issue due to dissolution of halite)	"The sediments consist of alternating shale, siltstone, sandstone, and gravel that were deposited in a variety of fluvial, lacustrine, and deltaic environments (Bradley and Kalaswad, 2003). The Dockum Group constitutes the hydrostratigraphic unit for the Dockum Aquifer which is classified as a minor aquifer in Texas (Ashworth and Hopkins, 1995; George and others, 2011)." (Meyer et al., 2012)
Rustler formation	Clastic sedimentary rock aquifer (consolidated or semi-consolidated rock) (water quality issue due to dissolution of halite)	"The Rustler Formation consists of Upper Permian sediments deposited unconformably on the Permian Salado Formation. The sediments consist of shale, silt, sandstone, dolomite, and the evaporites, halite and gypsum (anhydrite at depth). The Rustler Formation constitutes the hydrostratigraphic unit for the Rustler Aquifer, although the areal extent of the formation is much larger than the mapped limits of the aquifer." (Meyer et al., 2012)

Formation name	Category	Quote
Salado and Castile Formation	Carbonate aquifer (Evaporite exists)	"Capitan Reef Complex Aquifer". (Meyer et al., 2012). "The structurally elevated ridge separating the Pecos and Monument Draw troughs is underlain by the thickest sections of evaporites of the Salado and Castile formations". (Meyer et al., 2012)

Meyer, J.E., Wise, M.R., Kalaswad, S. (2012). *Pecos Valley aquifer, west Texas: structure and brackish groundwater*. Texas Water Development Board Report.

3.11 Trinity Aquifer System, Edwards-Trinity Aquifer System

Supplementary Fig. 119. Hydrogeologic cross section. 20 equally spaced transparent pink bars overlie the cross section; each shaded bar depicts the vertical offset from the land surface to the top of the uppermost confining unit or endogenous bedrock.

Supplementary Fig. 120. Vertical variations in the prevalence of wells that have been defined as tapping an unconfined or a confined aquifer by the USGS. The smaller squares represent 10 m depth intervals from the land surface to 100 m; the larger squares represent 20 m intervals from 100 m to 300 m below the land surface.

The Trinity Aquifer System is located in central Texas.

(i) A hydrogeologic cross section presented in Fig. 6-38 by Bruun et al. (2016) depicts layered sequences of sedimentary aquifers and aquitards. The uppermost Washita and Fredericksburg Formations are shown as aquitards, but may contain permeable portions that serve as aquifers (see (iii) below).

(ii) We analysed wells within the study area that the USGS has defined as either unconfined or confined. Most (>80%) wells at depths of 90-100 m and at depths exceeding 90 m are defined as tapping a confined aquifer.

(iii) Bruun et al. (2016) state (quote) "Sand distribution and thickness largely controls the productivity of the aquifer. The depositional environment in the Cretaceous Period resulted in a layered system of aquifers and aquitards in the northern Trinity Group." Mace et al. (1994) state (quote) "Hydrologic parameters for the Washita and Fredericksburg Groups... were estimated on the basis of rock type. These units are composed of approximately 40 percent shale and 60 percent limestone..."

Depth to confined conditions: 90-100 meters below land surface (based on (ii) above)

References: Bruun, B., Jackson, K., Lake, P., Walker, J. (2016). Texas Aquifers Study. Texas Water Development Board Report. 336 pp. Accessed April 1, 2021 from https://www.twdb.texas.gov/groundwater/docs/studies/TexasAquifersStudy_2016.pdf#page=89

Mace, R.E., Nance, H.S., Dutton, A.R. (1996). Geologic and Hydrogeologic Framework of Regional Aquifers in the Twin Mountains, Paluxy, and Woodbine Formations near the SSC Site, North-Central Texas. Topical Report to Texas National Research Laboratory. <https://www.beg.utexas.edu/files/publications/contract-reports/CR1994-Mace-1.pdf>

The table below presents a series of published quotes (see quotation marks denoting text quoted from another publication, which is cited following the quotation marks with the full reference written in full below the table). The leftmost column lists a title of a hydrogeologic formation depicted in the cross section on the previous page. The rightmost column presents a quote from a hydrogeological study (see base of table for citation). The quote has been annotated with colored text to highlight how we categorized each layer (i.e., see categories in the center column in the table). Specifically: (i) blue text highlights portions of a quote that provide insights into the degree of consolidation of the formation, (ii) red text highlights portions of a quote that categorize the formation as an aquifer or an aquitard (i.e., higher versus lower permeability in the context of local hydrogeologic formations), and (iii) green text highlights portions of a quote that provide information about the lithology of the formation.

Supplementary Table of 14. Hydrostratigraphy details for the Trinity Aquifer System

Formation name	Category	Quote
Younger Formation	Unconfined aquifer	"Sand, silt, clay and gravel." (Nordstrom, 1982). "Yields small to large amount of water to wells along the red river." (Nordstrom, 1982).
Woodbine group	Sedimentary rock aquifer (consolidated or semi-consolidated rock)	"The Woodbine Group is the only important aquifer of the Gulf Series in the area covered by this report. It consists of sand, sandstone, and clay and is capable of yielding small to large amounts of water." (Nordstrom, 1982).
Washita and Fredericksburg Groups	Sedimentary rock aquitard (consolidated or semi-consolidated rock)	"Both the Washita and Fredericksburg Groups of the Comanche Series consist predominantly of limestone, shale, clay, and marl and yield only small amounts of water to localized areas." (Nordstrom, 1982).
Paluxy Formation	Sedimentary rock aquifer (consolidated or semi-consolidated rock)	"The Paluxy is composed predominantly of fine- to coarse-grained, friable, homogeneous, white quartz sand interbedded with sandy, silty, calcareous, or waxy clay and shale. In general, coarse-grained sand is in the lower part." (Nordstrom, 1982). "The Paluxy Formation is an important aquifer in the study region and during 1974, produced over 10,000 acre-feet (12.3 hm ³) of water for municipal and industrial use and provided water to many domestic and livestock wells." (Nordstrom, 1982).
Glen Rose Formation	Sedimentary rock aquitard (consolidated or semi-consolidated rock)	"The Glen Rose is predominantly a limestone and yields small quantities of water only to localized areas." (Nordstrom, 1982). "A middle section of Antlers contains considerably more clay beds than the upper or lower sections, and to the south, near the updip limit of the Glen Rose Formation, limestone beds also occur." (Nordstrom, 1982).
Travis Peak/Twin Mountain Formation includes: (Hensell, Pearsall Member, & Hossion Member)	Sedimentary rock aquifer (consolidated or semi-consolidated rock)	"Originally the basal Cretaceous bed was named the Travis Peak Formation, but the name was changed to the Twin Mountains Formation in north-central Texas (Fisher and Rodda, 1966)." (Nordstrom, 1982). "The Twin Mountains consists of a basal conglomerate of chert and quartz, grading upward into coarse- to fine-grained sand interspersed with varicolored shale." (Nordstrom, 1982). "The Twin Mountains Formation is the most important source of ground water for a large part of the study region and yields moderate to large quantities of fresh to slightly

Formatted: Strong, Font: 10 pt, Not Bold, Font color: Text 1,

Formatted: Normal

Formatted: Font: 10 pt,

Formation name	Category	Quote
		saline water to municipal and industrial wells." (Nordstrom, 1982).

Nordstrom, P.L., 1982, Occurrence, availability, and chemical quality of ground water in the Cretaceous aquifers of northcentral Texas: Texas Department of Water Resources Report 269, v. 1, 109 p., and v. 2, 387 p. Accessed on June 15, 2022 via https://www.twdb.texas.gov/publications/reports/numbered_reports/doc/R269/R269v1/R269v1.pdf

3.12 Dougherty Plain and Marianna Lowlands, Floridan Aquifer System

Supplementary Fig. 121. Hydrogeologic cross section. 20 equally spaced transparent pink bars overlie the cross section; each shaded bar depicts the vertical offset from the land surface to the top of the uppermost confining unit or endogenous bedrock.

Supplementary Fig. 122. Vertical variations in the prevalence of wells that have been defined as tapping an unconfined or a confined aquifer by the USGS. The smaller squares represent 10 m depth intervals from the land surface to 100 m; the larger squares represent 20 m intervals from 100 m to 300 m below the land surface.

The Dougherty Plain and Marianna Lowlands are located in southwestern Georgia, southeastern Alabama, and western Florida.

(i) A hydrogeologic cross section presented in Plate 8 by Williams and Kuniansky (2016) depicts a shallow confining unit (Hawthorn Group) underlain by carbonate aquifers and confining units.

(ii) We analysed wells within the study area that the USGS has defined as either unconfined or confined. Most (>80%) wells at depths of 30-40 m and at depths exceeding 30 m are defined as tapping a confined aquifer.

Depth to confined conditions: 30-40 meters below land surface (based on (ii) above)

Reference: Williams, L.J., Kuniansky, E.L. (2016). Revised hydrogeologic framework of the Floridan aquifer system in Florida and parts of Georgia, Alabama, and South Carolina. US Geological Survey Professional Paper 1807, 140 pp. Accessed March 31, 2021 from <https://pubs.usgs.gov/pp/1807/pdf/pp1807.pdf>

The table below presents a series of published quotes (see quotation marks denoting text quoted from another publication, which is cited following the quotation marks with the full reference written in full below the table). The leftmost column lists a title of a hydrogeologic formation depicted in the cross section on the previous page. The rightmost column presents a quote from a hydrogeological study (see base of table for citation). The quote has been annotated with colored text to highlight how we categorized each layer (i.e., see categories in the center column in the table). Specifically: (i) blue text highlights portions of a quote that provide insights into the degree of consolidation of the formation, (ii) red text highlights portions of a quote that categorize the formation as an aquifer or an aquitard (i.e., higher versus lower permeability in the context of local hydrogeologic formations), and (iii) green text highlights portions of a quote that provide information about the lithology of the formation.

Supplementary Table 15. Hydrostratigraphy details for the Dougherty Plain and Marianna Lowlands

Formation name	Category	Quote
Undifferentiated Sand, Silt and Clay	Unconsolidated aquifer	"The surficial aquifer system consists mostly of sand and locally contains gravel and sandy lime stone of Pliocene to Holocene age. Where these sediments are thick and highly permeable, " (Williams, and Kuniansky, 2016). "The surficial aquifer system forms a thin irregular blanket of terrace and alluvial sands that can act as an important source sink layer for temporary storage of groundwater that may ultimately recharge the underlying Floridan aquifer system. " (Williams, and Kuniansky, 2016)
Hawthorn Group	Sedimentary rock aquitard (consolidated or semi-consolidated rock)	"The thickest and most extensive Miocene unit in the study area is the Hawthorn Group . The lithology of the Hawthorn varies greatly between areas, but mostly consists of phosphatic clay, silt, and sand that range in color from cream or gray to green to brown. " (Williams and Kuniansky, 2016). "The entire formation forms a thick, generally clastic, highly variable sequence of lower permeability rock that, where present, is considered to be the upper confining unit of the Floridan aquifer system. " (Williams and Kuniansky, 2016).
Upper Floridan aquifer	Carbonate aquifer	"The Upper Floridan aquifer includes the uppermost or shallowest permeable zones in the Floridan aquifer system." (Williams, and Kuniansky, 2016) "The Floridan aquifer system includes the vertically continuous carbonate-rock system described by Miller (1986)". (Williams, and Kuniansky, 2016). "the carbonate rocks of the Upper Floridan aquifer may directly overlie clastic rocks of the Lisbon and Claiborne aquifers;" (Williams, and Kuniansky, 2016). "Over a large part of the Floridan aquifer system, the top is marked by Oligocene rocks (Suwannee Limestone or equivalent) where such rocks are permeable and in hydraulic connection with the main part of the system. " (Williams and Kuniansky, 2016). "the Upper Floridan aquifer consists mostly of Oligocene Suwannee Limestone (if present) and late-Eocene Ocala Limestone. " (Williams and Kuniansky, 2016).
Lisbon-McBean confining unit	Sedimentary rock aquifer (consolidated or semi-consolidated rock)	"The composite unit also has been identified by Faye and Mayer (1997) as the Lisbon-McBean confining unit. " (Williams and Kuniansky, 2016). Table 7 of (Williams and Kuniansky, 2016), under the "Equivalent hydrogeologic unit": "Lisbon-McBean confining unit" and "Lithology" as " Clay,

Formation name	Category	Quote
		sand, argillaceous limestone ", under the "Water-bearing properites" " Mostly confining , may vary laterally depend on lithology".
Lower Floridan aquifer	Carbonate aquifer	"The Lower Floridan aquifer includes all permeable and less-permeable zones below (1) the MAPCU in peninsular Florida, (2) the BCCU in the western Florida panhandle and contiguous areas in southwestern Alabama, and (3) the LISAPCU in southeastern Alabama, Georgia, western South Carolina and northern Florida." (Williams and Kuniansky, 2016).
Lower confining unit	Sedimentary rock aquitard (consolidated or semi-consolidated rock)	"The Floridan is underlain everywhere by low-permeability rocks called the lower confining unit , which separates the Floridan aquifer system from older, deeper aquifers of the Southeastern Coastal Plain aquifer system." (Williams and Kuniansky, 2016). "The base of the Floridan aquifer system is marked by the lower confining unit , consisting of predominantly low-permeability late Paleocene to middle Eocene rocks ." (Williams and Kuniansky, 2016).
Clayton aquifer (sometimes, sand can dominate the limestone presence)	Carbonate aquifer	"The limestone of the Clayton aquifer is not hydraulically connected to other Tertiary limestones of the Floridan aquifer system." (Williams and Kuniansky, 2016). "In western Georgia, the uppermost part of the Clayton is a hard, sandy, fossiliferous limestone that constitutes an important aquifer (Clarke and others, 1984)." (Williams and Kuniansky, 2016).
Paleocene rocks	Carbonate aquifer	" Paleocene rocks in the study area can be categorized into three groups. The first group, which is the most important to this study, primarily consists of interbedded dolomite and anhydrite of the Cedar Keys Formation , which underlies part of southeastern Georgia and all of peninsular Florida." (Williams and Kuniansky, 2016). " Interbedded dolomite and anhydrite, both being of lower permeability , compose the lower two-thirds of the Cedar Keys Formation of peninsular Florida and southeastern Georgia. This carbonate-evaporite sequence forms the base of the Floridan aquifer system ." (Williams and Kuniansky, 2016).
Cretaceous formation	Sedimentary rock aquitard (consolidated or semi-consolidated rock)	"South of Brunswick, Ga., the base of the system drops several hundred feet into late Cretaceous rocks consisting of soft, friable, possibly Tayloran age, limestone with the permeable, late Cretaceous, Navarroan age Lawson Limestone above included as part of the Lower Floridan aquifer (Miller, 1986)." (Williams and Kuniansky, 2016). Figure 2 of (Williams and Kuniansky, 2016) under "Hydrogeology" in "Florida" " Confining unit ". (William, Raines, Lanning, 2016) under the "Hydrogeology" in "Florida" " confining unit ".

Williams, L.J., and Kuniansky, E.L. (2016). Revised hydrogeologic framework of the Floridan aquifer system in Florida and parts of Georgia, Alabama, and South Carolina (ver. 1.1, March 2016); U.S. Geological Survey Professional Paper 1807, 140 p., available at <https://pubs.usgs.gov/ds/760/>.

Williams, L.J., Raines, J.E., and Lanning, A.E. (2016). Geophysical log database for the Floridan aquifer system and southeastern Coastal Plain aquifer system in Florida and parts of Georgia, Alabama, and South Carolina (ver. 1.1, December 2016); U.S. Geological Survey Data Series 760, 12 p., available at <http://pubs.usgs.gov/ds/760/>

3.13 Eastern Flatwoods Southshores, Floridan Aquifer System

Supplementary Fig. 123. Hydrogeologic cross section. 20 equally spaced transparent pink bars overlie the cross section; each shaded bar depicts the vertical offset from the land surface to the top of the uppermost confining unit or endogenous bedrock.

Supplementary Fig. 124. Vertical variations in the prevalence of wells that have been defined as tapping an unconfined or a confined aquifer by the USGS. The smaller squares represent 10 m depth intervals from the land surface to 100 m; the larger squares represent 20 m intervals from 100 m to 300 m below the land surface.

The Eastern Flatwoods Southshores is located in southeastern Florida.

(i) A hydrogeologic cross section presented in Plate 2 by Williams and Kuniansky (2016) shows undifferentiated sand, silt and clay (~70 m thick) underlain by the Hawthorn Group confining unit.

(ii) We analysed wells within the study area that the USGS has defined as either unconfined or confined. Most (>80%) wells at depths of 100-120 m and at depths exceeding 100 m are defined as tapping a confined aquifer.

Depth to confined conditions: 100-120 meters below land surface (based on (ii) above)

Reference: Williams, L.J., Kuniansky, E.L. (2016). Revised hydrogeologic framework of the Floridan aquifer system in Florida and parts of Georgia, Alabama, and South Carolina. US Geological Survey Professional Paper 1807, 140 pp. Accessed March 31, 2021 from <https://pubs.usgs.gov/pp/1807/pdf/pp1807.pdf>

The table below presents a series of published quotes (see quotation marks denoting text quoted from another publication, which is cited following the quotation marks with the full reference written in full below the table). The leftmost column lists a title of a hydrogeologic formation depicted in the cross section on the previous page. The rightmost column presents a quote from a hydrogeological study (see base of table for citation). The quote has been annotated with colored text to highlight how we categorized each layer (i.e., see categories in the center column in the table). Specifically: (i) blue text highlights portions of a quote that provide insights into the degree of consolidation of the formation, (ii) red text highlights portions of a quote that categorize the formation as an aquifer or an aquitard (i.e., higher versus lower permeability in the context of local hydrogeologic formations), and (iii) green text highlights portions of a quote that provide information about the lithology of the formation.

Supplementary Table 16. Hydrostratigraphy details for the Eastern Flatwoods Southshores

Formation name	Category	Quote
Undifferentiated Sand, Silt and Clay	Unconsolidated aquifer	"The surficial aquifer system consists mostly of sand and locally contains gravel and sandy lime stone of Pliocene to Holocene age. Where these sediments are thick and highly permeable ," (Williams, and Kuniansky, 2016). "The surficial aquifer system forms a thin irregular blanket of terrace and alluvial sands that can act as an important source sink layer for temporary storage of groundwater that may ultimately recharge the underlying Floridan aquifer system ," (Williams, and Kuniansky, 2016)
Upper confining unit (Hawthorn Group)	Sedimentary rock aquitard (consolidated or semi-consolidated rock)	"The thickest and most extensive Miocene unit in the study area is the Hawthorn Group . The lithology of the Hawthorn varies greatly between areas, but mostly consists of phosphatic clay, silt, and sand that range in color from cream or gray to green to brown ," (Williams and Kuniansky, 2016). "The entire formation forms a thick, generally clastic, highly variable sequence of lower permeability rock that, where present, is considered to be the upper confining unit of the Floridan aquifer system ," (Williams and Kuniansky, 2016).
Top of Floridan aquifer (Suwannee and Ocala limestone)	Carbonate aquifer	"The Upper Floridan aquifer includes the uppermost or shallowest permeable zones in the Floridan aquifer system," (Williams, and Kuniansky, 2016) "The Floridan aquifer system includes the vertically continuous carbonate-rock system described by Miller (1986)". (Williams, and Kuniansky, 2016). "the carbonate rocks of the Upper Floridan aquifer may directly overlie clastic rocks of the Lisbon and Claiborne aquifers;" (Williams, and Kuniansky, 2016). "Over a large part of the Floridan aquifer system, the top is marked by Oligocene rocks (Suwannee Limestone or equivalent) where such rocks are permeable and in hydraulic connection with the main part of the system ," (Williams and Kuniansky, 2016). "the Upper Floridan aquifer consists mostly of Oligocene Suwannee Limestone (if present) and late-Eocene Ocala Limestone ," (Williams and Kuniansky, 2016).
Ocala-Avon Park lower permeability zone (layers of semiconfined unit exists)	Sedimentary rock aquitard (consolidated or semi-consolidated rock)	"The aggregate Avon Park permeable zone is overlain everywhere by a less-permeable carbonate zone named by others the " Ocala-Avon Park lower permeability zone " (a lower permeability zone within the Upper Floridan aquifer) and underlain by lower permeability confining to semiconfined evaporitic and non-evaporitic rocks of the newly

Formation name	Category	Quote
		mapped middle Avon Park composite unit (previously all or parts of MCUI–III and VI).” (Williams and Kuniansky, 2016).
Aggregate Avon Park permeable zone (Upper dolomite sand unit is leaky aquifer)	Carbonate aquifer	“Within the Upper Floridan aquifer of central and southern Florida, a subregionally extensive, highly fractured and cavernous interval called the Avon Park permeable zone is mapped as an aggregate of several permeable zones in the upper part of the Avon Park Formation. ” (Williams and Kuniansky, 2016).
Middle Avon park composite unit (semiconfining)	Sedimentary rock aquitard (consolidated or semi-consolidated rock)	“The aggregate Avon Park permeable zone is overlain everywhere by a less-permeable carbonate zone named by others the “Ocala-Avon Park lower permeability zone” (a lower permeability zone within the Upper Floridan aquifer) and underlain by lower permeability confining to semiconfining evaporitic and non-evaporitic rocks of the newly mapped middle Avon Park composite unit (previously all or parts of MCUI–III and VI). ” (Williams and Kuniansky, 2016).
Lower Avon Park permeable zone (this aquifer unit has thin layer of semiconfining unit).	Carbonate aquifer	“Below the MAPCU, permeable zones in the lower Avon Park Formation form the lower Avon Park permeable zone (LAPPZ) . The LAPPZ is mapped as a relatively thick zone of higher permeability with some lower permeability rocks that lie between the MAPCU and the glauconite marker unit. Freshwater parts of the LAPPZ are used for water supply in east-central Florida, where the zone is part of the Lower Floridan aquifer. ” (Williams and Kuniansky, 2016).
Glauconite marker unit	Sedimentary rock aquitard (consolidated or semi-consolidated rock)	“ Glauconite Marker Unit: Peninsular Florida —Over much of peninsular Florida, a distinctive lower permeability unit lies deep within the Floridan aquifer system near its base. ” (Williams and Kuniansky, 2016).

Williams, L.J., Kuniansky, E.L. (2016). Revised hydrogeologic framework of the Floridan aquifer system in Florida and parts of Georgia, Alabama, and South Carolina (ver. 1.1, March 2016): U.S. Geological Survey Professional Paper 1807, 140 p., 23 pls., <http://dx.doi.org/10.3133/pp1807>

3.14 Lower Coastal Plain, Floridan Aquifer System

Supplementary Fig. 125. Hydrogeologic cross section. 20 equally spaced transparent pink bars overlie the cross section; each shaded bar depicts the vertical offset from the land surface to the top of the uppermost confining unit or endogenous bedrock.

Supplementary Fig. 126. Vertical variations in the prevalence of wells that have been defined as tapping an unconfined or a confined aquifer by the USGS. The smaller squares represent 10 m depth intervals from the land surface to 100 m; the larger squares represent 20 m intervals from 100 m to 300 m below the land surface.

The Lower Coastal Plain is located in the southern coastal region of South Carolina.

(i) A hydrogeologic cross section presented in Fig. 11 by Aucott (1996) depicts a confining unit at depths of ~80 to ~150 m below the land surface.

(ii) We analysed wells within the study area that the USGS has defined as either unconfined or confined. Most (>80%) wells at depths of 90-100 m and at depths exceeding 90 m are defined as tapping a confined aquifer.

Reference: Aucott, W.R. (1996). Hydrology of the Southeastern Coastal Plain Aquifer System in South Carolina and Parts of Georgia and North Carolina. U.S Geological Survey Professional Paper 1410-E, 95 pp. Accessed March 31, 2021 from <https://pubs.usgs.gov/pp/1410e/report.pdf>

The table below presents a series of published quotes (see quotation marks denoting text quoted from another publication, which is cited following the quotation marks with the full reference written in full below the table). The leftmost column lists a title of a hydrogeologic formation depicted in the cross section on the previous page. The rightmost column presents a quote from a hydrogeological study (see base of table for citation). The quote has been annotated with colored text to highlight how we categorized each layer (i.e., see categories in the center column in the table). Specifically: (i) blue text highlights portions of a quote that provide insights into the degree of consolidation of the formation, (ii) red text highlights portions of a quote that categorize the formation as an aquifer or an aquitard (i.e., higher versus lower permeability in the context of local hydrogeologic formations), and (iii) green text highlights portions of a quote that provide information about the lithology of the formation.

Supplementary Table 17. Hydrostratigraphy details for the Lower Coastal Plain

Formation name	Category	Quote
Surficial aquifer	Unconsolidated aquifer	"The surficial aquifer consists of marine terrace deposits . These sediments are generally less than 40 ft thick and consist primarily of sand, shell, and clay that were deposited during a series of transgressions and regressions of the sea during the Pleistocene Epoch (Siple, 1946)." (Aucott, 1999).
Floridan aquifer system	Carbonate aquifer	"The Floridan aquifer system in South Carolina generally consists of white to creamy-yellow limestone of late to middle Eocene age. The sediments composing this system are parts of the Cooper Group, the Ocala Limestone (where present), and the underlying Santee Limestone (table 1)." (Aucott, 1999).
Tertiary Sand aquifer	Sedimentary rock aquifer (consolidated or semi-consolidated rock)	"The Tertiary sand aquifer partly underlies the carbonate rocks composing the Floridan aquifer system and partly is the clastic facies equivalent of the carbonate rocks." (Aucott, 1999). " Sediments composing this aquifer include the Barnwell, McBean, and Congaree Formations and the upper part of the Black Mingo Formation (table 1); they consist of fine to medium sand and clay, commonly light greenish yellow to orange ." (Aucott, 1999).
Confining unit	Sedimentary rock aquitard (consolidated or semi-consolidated rock)	"Vertical movement across a confining unit is usually controlled by the least permeable layer within the confining unit, which is typically a tight, marine clay in the Coastal Plain sediments of South Carolina. All of the confining units identified allow limited vertical movement of water through them ." (Aucott, 1999).
Black Creek aquifer	Sedimentary rock aquifer (consolidated or semi-consolidated rock)	" Sediments composing the Black Creek Formation are principally thin, laminated layers of gray, fine to medium, micaceous sand and dark-gray to black clay . The coarseness of the sands and the clay content vary areally . The Black Creek aquifer is the uppermost regional aquifer consisting of sediments of Cretaceous age." (Aucott, 1999).
Confining unit	Sedimentary rock aquitard (consolidated or semi-consolidated rock)	"The confining unit between the Black Creek aquifer and the Middendorf aquifer primarily consists of sandy clay in the lower part of the Black Creek Formation." (Aucott, 1999).
Middendorf aquifer	Sedimentary rock aquifer (consolidated or	"In the lower Coastal Plain, the sediments of the Middendorf aquifer are lithologically similar to those of the Black Creek

Formation name	Category	Quote
	semi-consolidated rock	aquifer and consist of thin, laminated layers of fine to medium sand and clay. " (Aucott, 1999).
Confining unit	Sedimentary rock aquitard (consolidated or semi-consolidated rock)	"The confining unit between the Middendorf and Cape Fear aquifers is very effective in separating the flow systems of these aquifers in the eastern part of the study area." (Aucott, 1999).
Cape Fear aquifer	Sedimentary rock aquifer (consolidated or semi-consolidated rock)	"The Cape Fear aquifer consists of the lower part of the Cape Fear Formation and is the basal aquifer in the Coastal Plain aquifer system of South Carolina." (Aucott, 1999). "The Cape Fear aquifer consists predominantly of sand, silt, and gravel separated by relatively thick silt and clay layers." (Aucott, 1999).
Rocks of pre-Cretaceous age	Sedimentary rock aquitard (consolidated or semi-consolidated rock)	"Underlying the sediments that compose the Coastal Plain aquifer system are pre-Cretaceous igneous, metamorphic, and consolidated sedimentary rocks, all of low permeability. The flow of water within these pre-Cretaceous rocks and between them and the overlying Coastal Plain aquifers is considerably less than flow within the Coastal Plain aquifers." (Aucott, 1999).

Aucott, W.R. (1996). Hydrology of the Southeastern Coastal Plain Aquifer System in South Carolina and Parts of Georgia and North Carolina. U.S Geological Survey Professional Paper 1410-E, 95 pp. Accessed March 31, 2021 from <https://pubs.usgs.gov/pp/1410e/report.pdf>

3.15 Ocala Uplift, Floridan Aquifer System

Supplementary Fig. 127. Hydrogeologic cross section. 20 equally spaced transparent pink bars overlie the cross section; each shaded bar depicts the vertical offset from the land surface to the top of the uppermost confining unit or endogenous bedrock.

Supplementary Fig. 128. Vertical variations in the prevalence of wells that have been defined as tapping an unconfined or a confined aquifer by the USGS. The smaller squares represent 10 m depth intervals from the land surface to 100 m; the larger squares represent 20 m intervals from 100 m to 300 m below the land surface.

The Ocala Uplift portion of the Floridan Aquifer System is located in western Florida.

(i) A hydrogeologic cross section presented in Plate 2 by Williams and Kuniansky (2016) depicts the Suwannee and Ocala limestone units near the land surface but, in some places, it is overlain by a shallow confining unit (the Hawthorn Group).

(ii) We analysed wells within the study area that the USGS has defined as either unconfined or confined. Most (>80%) wells at depths of 30-40 m and at depths exceeding 30 m are defined as tapping a confined aquifer.

Depth to confined conditions: 30-40 meters below land surface (based on (ii) above)

Reference: Williams, L.J., Kuniansky, E.L. (2016). Revised hydrogeologic framework of the Floridan aquifer system in Florida and parts of Georgia, Alabama, and South Carolina. US Geological Survey Professional Paper 1807, 140 pp. Accessed March 31, 2021 from <https://pubs.usgs.gov/pp/1807/pdf/pp1807.pdf>

The table below presents a series of published quotes (see quotation marks denoting text quoted from another publication, which is cited following the quotation marks with the full reference written in full below the table). The leftmost column lists a title of a hydrogeologic formation depicted in the cross section on the previous page. The rightmost column presents a quote from a hydrogeological study (see base of table for citation). The quote has been annotated with colored text to highlight how we categorized each layer (i.e., see categories in the center column in the table). Specifically: (i) blue text highlights portions of a quote that provide insights into the degree of consolidation of the formation, (ii) red text highlights portions of a quote that categorize the formation as an aquifer or an aquitard (i.e., higher versus lower permeability in the context of local hydrogeologic formations), and (iii) green text highlights portions of a quote that provide information about the lithology of the formation.

Supplementary Table 18. Hydrostratigraphy details for the Ocala Uplift

Formation name	Category	Quote
Undifferentiated Sand, Silt and Clay	Unconsolidated aquifer	"The surficial aquifer system consists mostly of sand and locally contains gravel and sandy lime stone of Pliocene to Holocene age. Where these sediments are thick and highly permeable ," (Williams, and Kuniansky, 2016). "The surficial aquifer system forms a thin irregular blanket of terrace and alluvial sands that can act as an important source sink layer for temporary storage of groundwater that may ultimately recharge the underlying Floridan aquifer system ." (Williams, and Kuniansky, 2016)
Upper confining unit (Hawthorn Group)	Sedimentary rock aquitard (consolidated or semi-consolidated rock)	"The thickest and most extensive Miocene unit in the study area is the Hawthorn Group . The lithology of the Hawthorn varies greatly between areas, but mostly consists of phosphatic clay, silt, and sand that range in color from cream or gray to green to brown ." (Williams and Kuniansky, 2016). "The entire formation forms a thick, generally clastic, highly variable sequence of lower permeability rock that, where present, is considered to be the upper confining unit of the Floridan aquifer system ." (Williams and Kuniansky, 2016).
Top of Floridan aquifer system (Suwannee and Ocala limestone)	Carbonate aquifer	"The Upper Floridan aquifer includes the uppermost or shallowest permeable zones in the Floridan aquifer system." (Williams, and Kuniansky, 2016) "The Floridan aquifer system includes the vertically continuous carbonate-rock system described by Miller (1986)". (Williams, and Kuniansky, 2016). "the carbonate rocks of the Upper Floridan aquifer may directly overlie clastic rocks of the Lisbon and Claiborne aquifers;" (Williams, and Kuniansky, 2016). "Over a large part of the Floridan aquifer system, the top is marked by Oligocene rocks (Suwannee Limestone or equivalent) where such rocks are permeable and in hydraulic connection with the main part of the system ." (Williams and Kuniansky, 2016). "the Upper Floridan aquifer consists mostly of Oligocene Suwannee Limestone (if present) and late-Eocene Ocala Limestone ." (Williams and Kuniansky, 2016).
Ocala-Avon Park lower permeability zone (layers of semiconfined unit exists)	Sedimentary rock aquitard (consolidated or semi-consolidated rock)	"The aggregate Avon Park permeable zone is overlain everywhere by a less-permeable carbonate zone named by others the " Ocala-Avon Park lower permeability zone " (a lower permeability zone within the Upper Floridan aquifer) and underlain by lower permeability confining to semiconfining evaporitic and non-evaporitic rocks of the newly

		mapped middle Avon Park composite unit (previously all or parts of MCUI–III and VI). " (Williams and Kuniansky, 2016).
Aggregate Avon Park permeable zone (Upper dolomite sand unit is leaky aquifer)	Carbonate aquifer	"Within the Upper Floridan aquifer of central and southern Florida, a subregionally extensive, highly fractured and cavernous interval called the Avon Park permeable zone is mapped as an aggregate of several permeable zones in the upper part of the Avon Park Formation." (Williams and Kuniansky, 2016).
Middle Avon Park composite unit (semiconfining)	Sedimentary rock aquitard (consolidated or semi-consolidated rock)	"The aggregate Avon Park permeable zone is overlain everywhere by a less-permeable carbonate zone named by others the "Ocala-Avon Park lower permeability zone" (a lower permeability zone within the Upper Floridan aquifer) and underlain by lower permeability confining to semiconfining evaporitic and non-evaporitic rocks of the newly mapped middle Avon Park composite unit (previously all or parts of MCUI–III and VI)." (Williams and Kuniansky, 2016).
Lower Avon Park permeable zone	Carbonate aquifer	"Below the MAPCU, permeable zones in the lower Avon Park Formation form the lower Avon Park permeable zone (LAPPZ). The LAPPZ is mapped as a relatively thick zone of higher permeability with some lower permeability rocks that lie between the MAPCU and the glauconite marker unit. Freshwater parts of the LAPPZ are used for water supply in east-central Florida, where the zone is part of the Lower Floridan aquifer." (Williams and Kuniansky, 2016).
Glauconite marker unit	Sedimentary rock aquitard (consolidated or semi-consolidated rock)	"Glauconite Marker Unit: Peninsular Florida—Over much of peninsular Florida, a distinctive lower permeability unit lies deep within the Floridan aquifer system near its base." (Williams and Kuniansky, 2016).
Oldsmar Formation	Carbonate aquifer	"Unit is in the Oldsmar Formation consisting almost entirely of yellowish brown, well indurated, cryptocrystalline to microcrystalline dolostone; in part calcareous, highly altered, vuggy with associated iron staining." (Williams and Kuniansky, 2016). "The Oldsmar permeable zone includes all extremely permeable zones and otherwise less permeable zones that lie between the glauconite marker unit and base of the massive dolostone unit in the Oldsmar Formation." (Williams and Kuniansky, 2016).
Low permeability rock near base of system	Sedimentary rock aquitard (consolidated or semi-consolidated rock)	"The Floridan is underlain everywhere by low-permeability rocks called the lower confining unit, which separates the Floridan aquifer system from older, deeper aquifers of the Southeastern Coastal Plain aquifer system." (Williams and Kuniansky, 2016). "The base of the Floridan aquifer system is marked by the lower confining unit, consisting of predominantly low-permeability late Paleocene to middle Eocene rocks." (Williams and Kuniansky, 2016).

Williams, L.J., and Kuniansky, E.L. (2016). Revised hydrogeologic framework of the Floridan aquifer system in Florida and parts of Georgia, Alabama, and South Carolina (ver. 1.1, March 2016): U.S. Geological Survey Professional Paper 1807, 140 p., 23 pls., <http://dx.doi.org/10.3133/pp1807>

3.16 Sea Island, Floridan Aquifer System

Supplementary Fig. 129. Hydrogeologic cross section. 20 equally spaced transparent pink bars overlie the cross section; each shaded bar depicts the vertical offset from the land surface to the top of the uppermost confining unit or endogenous bedrock.

Supplementary Fig. 130. Vertical variations in the prevalence of wells that have been defined as tapping an unconfined or a confined aquifer by the USGS. The smaller squares represent 10 m depth intervals from the land surface to 100 m; the larger squares represent 20 m intervals from 100 m to 300 m below the land surface.

The Sea Island area of the Floridan Aquifer System is located in eastern Georgia.

(i) A hydrogeologic cross section presented in Plate 1 by Williams and Kuniansky (2016) depicts a shallow layer of undifferentiated sand, silt and clay underlain by a confining unit.

(ii) We analysed wells within the study area that the USGS has defined as either unconfined or confined. Most (>80%) wells at depths of 30-40 m and at depths exceeding 30 m are defined as tapping a confined aquifer.

Depth to confined conditions: 30-40 meters below land surface (based on (ii) above)

Reference: Williams, L.J., Kuniansky, E.L. (2016). Revised hydrogeologic framework of the Floridan aquifer system in Florida and parts of Georgia, Alabama, and South Carolina. US Geological Survey Professional Paper 1807, 140 pp. Accessed March 31, 2021 from <https://pubs.usgs.gov/pp/1807/pdf/pp1807.pdf>

The table below presents a series of published quotes (see quotation marks denoting text quoted from another publication, which is cited following the quotation marks with the full reference written in full below the table). The leftmost column lists a title of a hydrogeologic formation depicted in the cross section on the previous page. The rightmost column presents a quote from a hydrogeological study (see base of table for citation). The quote has been annotated with colored text to highlight how we categorized each layer (i.e., see categories in the center column in the table). Specifically: (i) blue text highlights portions of a quote that provide insights into the degree of consolidation of the formation, (ii) red text highlights portions of a quote that categorize the formation as an aquifer or an aquitard (i.e., higher versus lower permeability in the context of local hydrogeologic formations), and (iii) green text highlights portions of a quote that provide information about the lithology of the formation.

Supplementary Table 19. Hydrostratigraphy details for the Sea Island

Formation name	Category	Quote
Undifferentiated Sand, Silt and clay	Unconsolidated aquifer	"The surficial aquifer system consists mostly of sand and locally contains gravel and sandy lime stone of Pliocene to Holocene age. Where these sediments are thick and highly permeable ," (Williams, and Kuniansky, 2016). "The surficial aquifer system forms a thin irregular blanket of terrace and alluvial sands that can act as an important source sink layer for temporary storage of groundwater that may ultimately recharge the underlying Floridan aquifer system ," (Williams, and Kuniansky, 2016)
Upper confining unit	Sedimentary rock aquitard (consolidated or semi-consolidated rock)	"The thickest and most extensive Miocene unit in the study area is the Hawthorn Group . The lithology of the Hawthorn varies greatly between areas, but mostly consists of phosphatic clay, silt, and sand that range in color from cream or gray to green to brown ," (Williams and Kuniansky, 2016). "The entire formation forms a thick, generally clastic, highly variable sequence of lower permeability rock that, where present, is considered to be the upper confining unit of the Floridan aquifer system ," (Williams and Kuniansky, 2016).
Upper Floridan aquifer	Carbonate aquifer	"The Upper Floridan aquifer includes the uppermost or shallowest permeable zones in the Floridan aquifer system," (Williams, and Kuniansky, 2016) "The Floridan aquifer system includes the vertically continuous carbonate-rock system described by Miller (1986)". (Williams, and Kuniansky, 2016). "the carbonate rocks of the Upper Floridan aquifer may directly overlie clastic rocks of the Lisbon and Claiborne aquifers;" (Williams, and Kuniansky, 2016).
Lisbon-Avon Park composite unit (A leaky to non-leaky(?), semiconfining unit locally gypsiferous limestone, dolomitic limestone, & dolostone in the middle part of the Avon Park Formation)	Sedimentary rock aquitard (consolidated or semi-consolidated rock)	"As shown in figure 6, the Upper and Lower Floridan aquifers are separated by the discontinuous numbered middle confining units that are now grouped within two composite units: (1) the Lisbon-Avon Park composite unit (LISAPCU) , consisting of lower permeability clastic and higher to lower permeability carbonate rocks in the northern part of the study area; and (2) the middle Avon Park composite unit (MAPCU) , consisting of lower permeability evaporite- and non-evaporite-bearing carbonate rocks to moderately permeable carbonate rocks in the central and southern parts of the study area." (Williams and Kuniansky, 2016).

Formation name	Category	Quote
Avon Park Formation	Carbonate aquifer	"In the northern coastal areas of Georgia and South Carolina, the strata that compose the Lower Floridan aquifer include limestone, dolomitic limestone, and dolomite that lie within the middle to lower part of the Avon Park Formation or equivalent middle Eocene formations (pl. 2). " (Williams and Kuniansky, 2016).
Middle Avon Park composite unit	Sedimentary rock aquitard (consolidated or semi-consolidated rock)	" Middle Avon Park Composite Unit, Central and Southern Florida —The MAPCU consists of lower permeability rocks in both evaporitic and non-evaporitic facies within the middle (or approximate middle) part of the Avon Park Formation (table 8, figs. 37 and 38). Because its permeability is generally lower than that of other less-permeable zones within the Floridan aquifer system, the MAPCU is considered the principal confining to semiconfining unit in peninsular Florida (fig. 38)." (Williams and Kuniansky, 2016).
Lower Avon Park permeable zone	Carbonate aquifer	"Below the MAPCU, permeable zones in the lower Avon Park Formation form the lower Avon Park permeable zone (LAPPZ) . The LAPPZ is mapped as a relatively thick zone of higher permeability with some lower permeability rocks that lie between the MAPCU and the glauconite marker unit. Freshwater parts of the LAPPZ are used for water supply in east-central Florida, where the zone is part of the Lower Floridan aquifer. " (Williams and Kuniansky, 2016).
Marl	Sedimentary rock aquitard (consolidated or semi-consolidated rock)	"In the northern coastal region of Georgia and South Carolina, the base of the Floridan aquifer system is formed by low-permeability middle Eocene marl (fig. 23)." (Williams and Kuniansky, 2016).
Oldsmar Formation	Carbonate aquifer	"Unit is in the Oldsmar Formation consisting almost entirely of yellowish brown, well indurated, cryptocrystalline to microcrystalline dolostone ; in part calcareous, highly altered, vuggy with associated iron staining." (Williams and Kuniansky, 2016). "The Oldsmar permeable zone includes all extremely permeable zones and otherwise less permeable zones that lie between the glauconite marker unit and base of the massive dolostone unit in the Oldsmar Formation. " (Williams and Kuniansky, 2016).
Upper Part of Fernandina permeable zone	Carbonate aquifer	"For example, the Fernandina permeable zone (Miller, 1986) is a cavernous interval within the Lower Floridan aquifer that has much higher hydraulic conductivity than the surrounding rock. " (Williams and Kuniansky, 2016).
Paleocene rock: Dublin-Midville aquifer	Carbonate aquifer	" Paleocene rocks in the study area can be categorized into three groups. The first group, which is the most important to this study, primarily consists of interbedded dolomite and anhydrite of the Cedar Keys Formation , which underlies part of southeastern Georgia and all of peninsular Florida." (Williams and Kuniansky, 2016). " Interbedded dolomite and anhydrite, both being of lower permeability, compose the lower two-thirds of the Cedar Keys Formation of peninsular Florida and southeastern Georgia. This carbonate-evaporite sequence forms the base of the Floridan aquifer system. " (Williams and Kuniansky, 2016).

Formation name	Category	Quote
Cretaceous Formation (Navarro rocks)	Sedimentary rock aquitard (consolidated or semi-consolidated rock)	"South of Brunswick, Ga., the base of the system drops several hundred feet into late Cretaceous rocks consisting of soft, friable, possibly Tayloran age, limestone with the permeable, late Cretaceous, Navarroan age Lawson Limestone above included as part of the Lower Floridan aquifer (Miller, 1986)." (Williams and Kuniansky, 2016). Figure 2 of (Williams and Kuniansky, 2016) under "Hydrogeology" in "Florida" "Confining unit". (William, Raines, Lanning, 2016) under the "Hydrogeology" in "Florida" "confining unit".

Williams, L.J., and Kuniansky, E.L. (2016). Revised hydrogeologic framework of the Floridan aquifer system in Florida and parts of Georgia, Alabama, and South Carolina (ver. 1.1, March 2016): U.S. Geological Survey Professional Paper 1807, 140 p., 23 pls., <http://dx.doi.org/10.3133/pp1807>

3.17 Tifton Upland, Floridan Aquifer System

Supplementary Fig. 131. Hydrogeologic cross section. 20 equally spaced transparent pink bars overlie the cross section; each shaded bar depicts the vertical offset from the land surface to the top of the uppermost confining unit or endogenous bedrock.

Supplementary Fig. 132. Vertical variations in the prevalence of wells that have been defined as tapping an unconfined or a confined aquifer by the USGS. The smaller squares represent 10 m depth intervals from the land surface to 100 m; the larger squares represent 20 m intervals from 100 m to 300 m below the land surface.

The Tifton Upland portion of the Floridan Aquifer System is located in south-central Georgia.

(i) A hydrogeologic cross section presented in Plate 8 by Williams and Kuniansky (2016) shows the Surficial Aquifer underlain by a relatively shallow confining unit (Hawthorn Group).

(ii) We analysed wells within the study area that the USGS has defined as either unconfined or confined. Most (>80%) wells at depths of 40-50 m and at depths exceeding 40 m are defined as tapping a confined aquifer.

Depth to confined conditions: 40-50 meters below land surface (based on (ii) above)

Reference: Williams, L.J., Kuniansky, E.L. (2016). Revised hydrogeologic framework of the Floridan aquifer system in Florida and parts of Georgia, Alabama, and South Carolina. US Geological Survey Professional Paper 1807, 140 pp. Accessed March 31, 2021 from <https://pubs.usgs.gov/pp/1807/pdf/pp1807.pdf>

The table below presents a series of published quotes (see quotation marks denoting text quoted from another publication, which is cited following the quotation marks with the full reference written in full below the table). The leftmost column lists a title of a hydrogeologic formation depicted in the cross section on the previous page. The rightmost column presents a quote from a hydrogeological study (see base of table for citation). The quote has been annotated with colored text to highlight how we categorized each layer (i.e., see categories in the center column in the table). Specifically: (i) blue text highlights portions of a quote that provide insights into the degree of consolidation of the formation, (ii) red text highlights portions of a quote that categorize the formation as an aquifer or an aquitard (i.e., higher versus lower permeability in the context of local hydrogeologic formations), and (iii) green text highlights portions of a quote that provide information about the lithology of the formation.

Supplementary Table 20. Hydrostratigraphy details for the Tifton Upland

Formation name	Category	Quote
Surficial aquifer system	Unconsolidated aquifer	"The surficial aquifer system consists mostly of sand and locally contains gravel and sandy lime stone of Pliocene to Holocene age. Where these sediments are thick and highly permeable ," (Williams, and Kuniansky, 2016). "The surficial aquifer system forms a thin irregular blanket of terrace and alluvial sands that can act as an important source sink layer for temporary storage of groundwater that may ultimately recharge the underlying Floridan aquifer system . (Williams, and Kuniansky, 2016)
Hawthorn Group	Sedimentary rock aquitard (consolidated or semi-consolidated rock)	"The thickest and most extensive Miocene unit in the study area is the Hawthorn Group . The lithology of the Hawthorn varies greatly between areas, but mostly consists of phosphatic clay, silt, and sand that range in color from cream or gray to green to brown ." (Williams and Kuniansky, 2016). "The entire formation forms a thick, generally clastic, highly variable sequence of lower permeability rock that, where present, is considered to be the upper confining unit of the Floridan aquifer system ." (Williams and Kuniansky, 2016).
Upper Floridan aquifer	Carbonate aquifer	"The Upper Floridan aquifer includes the uppermost or shallowest permeable zones in the Floridan aquifer system." (Williams, and Kuniansky, 2016) "The Floridan aquifer system includes the vertically continuous carbonate-rock system described by Miller (1986)". (Williams, and Kuniansky, 2016). "the carbonate rocks of the Upper Floridan aquifer may directly overlie clastic rocks of the Lisbon and Claiborne aquifers;" (Williams, and Kuniansky, 2016). "In Georgia, the Gulf Trough forms a narrow, distinct structural low at the top of the Floridan aquifer system (figs. 10 and 22). (Williams and Kuniansky, 2016).
Lisbon-McBean confining	Sedimentary rock aquitard (consolidated or semi-consolidated rock)	"The composite unit also has been identified by Faye and Mayer (1997) as the Lisbon-McBean confining unit ." (Williams and Kuniansky, 2016). Table 7 of (Williams and Kuniansky, 2016), under the "Equivalent hydrogeologic unit": "Lisbon-McBean confining unit" and "Lithology" as " Clay, sand, argillaceous limestone ", under the "Water-bearing properties" " Mostly confining , may vary laterally depend on lithology".

Formation name	Category	Quote
Claiborne aquifer	Carbonate aquifer	"In this area, and farther north toward the outcrop area, clastic rocks of the Claiborne aquifer (McFadden and Perriello, 1983) are hydraulically connected to the Upper Floridan aquifer. " (Williams and Kuniansky, 2016). " Sands of the Tallahatta and Huber Formations were included in the Gordon aquifer system (Brooks and others, 1985) and equivalent beds in southwestern Georgia were called the Claiborne aquifer (McFadden and Perriello, 1983). Like the Lisbon aquifer, the Gordon and Claiborne aquifers are considered part of the Floridan aquifer system in this report." (Williams and Kuniansky, 2016). "In Georgia, the Gordon aquifer consists of middle Eocene interbedded sand, silt, and clay and is equivalent to the Hatchetigbee and Tallahatta Formations and the lower part of the Lisbon Formation in western Georgia (Brooks and others, 1985)." (Williams and Kuniansky, 2016).
Lisbon-Avon Park composite unit (non-evaporitic), Middle Avon Park composite unit (some can be moderately permeable).	Sedimentary rock aquitard (consolidated or semi-consolidated rock)	"As shown in figure 6, the Upper and Lower Floridan aquifers are separated by the discontinuous numbered middle confining units that are now grouped within two composite units: (1) the Lisbon-Avon Park composite unit (LISAPCU), consisting of lower permeability clastic and higher to lower permeability carbonate rocks in the northern part of the study area; and (2) the middle Avon Park composite unit (MAPCU), consisting of lower permeability evaporite-and non-evaporite-bearing carbonate rocks to moderately permeable carbonate rocks in the central and southern parts of the study area." (Williams and Kuniansky, 2016).
Wilcox confining unit	Sedimentary rock aquitard (consolidated or semi-consolidated rock)	"In Georgia, the base is identified as the Wilcox confining zone , which separates the overlying Lower Floridan and Claiborne aquifers from the underlying Clayton aquifer (Clarke and others, 1984)." (Williams and Kuniansky, 2016). "The Floridan is underlain everywhere by low-permeability rocks called the lower confining unit , which separates the Floridan aquifer system from older, deeper aquifers of the Southeastern Coastal Plain aquifer system." (Williams and Kuniansky, 2016). "The base of the Floridan aquifer system is marked by the lower confining unit , consisting of predominantly low-permeability late Paleocene to middle Eocene rocks. " (Williams and Kuniansky, 2016).
Clayton aquifer (sometimes, sand can dominate the limestone presence)	Carbonate aquifer	"The limestone of the Clayton aquifer is not hydraulically connected to other Tertiary limestones of the Floridan aquifer system." (Williams and Kuniansky, 2016). "In western Georgia, the uppermost part of the Clayton is a hard, sandy, fossiliferous limestone that constitutes an important aquifer (Clarke and others, 1984)." (Williams and Kuniansky, 2016).
Cretaceous formation	Sedimentary rock aquitard (consolidated or semi-consolidated rock)	"South of Brunswick, Ga., the base of the system drops several hundred feet into late Cretaceous rocks consisting of soft, friable, possibly Tayloran age, limestone with the permeable, late Cretaceous, Navarroan age Lawson Limestone above included as

Formation name	Category	Quote
		part of the Lower Floridan aquifer (Miller, 1986)." (Williams and Kuniansky, 2016). Figure 2 of (Williams and Kuniansky, 2016) under "Hydrogeology" in "Florida" "Confining unit". (William, Raines, Lanning, 2016) under the "Hydrogeology" in "Florida" "confining unit".

Williams, L.J., and Kuniansky, E.L. (2016). Revised hydrogeologic framework of the Floridan aquifer system in Florida and parts of Georgia, Alabama, and South Carolina (ver. 1.1, March 2016): U.S. Geological Survey Professional Paper 1807, 140 p., 23 pls., <http://dx.doi.org/10.3133/pp1807>

3.18 Vidalia Upland, Floridan Aquifer System

Supplementary Fig. 133. Hydrogeologic cross section. 20 equally spaced transparent pink bars overlie the cross section; each shaded bar depicts the vertical offset from the land surface to the top of the uppermost confining unit or endogenous bedrock.

Supplementary Fig. 134. Vertical variations in the prevalence of wells that have been defined as tapping an unconfined or a confined aquifer by the USGS. The smaller squares represent 10 m depth intervals from the land surface to 100 m; the larger squares represent 20 m intervals from 100 m to 300 m below the land surface.

The Vidalia Upland area of the Floridan Aquifer System is located in central and eastern Georgia.

(i) A hydrogeologic cross section presented in Plate 8 by Williams and Kuniansky (2016) depicts undifferentiated sand, silt and clay (less than ~20 m thick) underlain by the Hawthorn Group confining unit.

(ii) We analysed wells within the study area that the USGS has defined as either unconfined or confined. Most (>80%) wells at depths of 10-20 m and at depths exceeding 10 m are defined as tapping a confined aquifer.

Depth to confined conditions: 10-20 meters below land surface (based on (ii) above)

Reference: Williams, L.J., Kuniansky, E.L. (2016). Revised hydrogeologic framework of the Floridan aquifer system in Florida and parts of Georgia, Alabama, and South Carolina. US Geological Survey Professional Paper 1807, 140 pp. Accessed March 31, 2021 from <https://pubs.usgs.gov/pp/1807/pdf/pp1807.pdf>

The table below presents a series of published quotes (see quotation marks denoting text quoted from another publication, which is cited following the quotation marks with the full reference written in full below the table). The leftmost column lists a title of a hydrogeologic formation depicted in the cross section on the previous page. The rightmost column presents a quote from a hydrogeological study (see base of table for citation). The quote has been annotated with colored text to highlight how we categorized each layer (i.e., see categories in the center column in the table). Specifically: (i) blue text highlights portions of a quote that provide insights into the degree of consolidation of the formation, (ii) red text highlights portions of a quote that categorize the formation as an aquifer or an aquitard (i.e., higher versus lower permeability in the context of local hydrogeologic formations), and (iii) green text highlights portions of a quote that provide information about the lithology of the formation.

Supplementary Table 21. Hydrostratigraphy details for the Vidalia Upland

Formation name	Category	Quote
Undifferentiated Sand, Silt and clay	Unconsolidated aquifer	"The surficial aquifer system consists mostly of sand and locally contains gravel and sandy lime stone of Pliocene to Holocene age. Where these sediments are thick and highly permeable ," (Williams, and Kuniansky, 2016). "The surficial aquifer system forms a thin irregular blanket of terrace and alluvial sands that can act as an important source sink layer for temporary storage of groundwater that may ultimately recharge the underlying Floridan aquifer system ." (Williams, and Kuniansky, 2016)
Upper confining unit	Sedimentary rock aquitard (consolidated or semi-consolidated rock)	"The thickest and most extensive Miocene unit in the study area is the Hawthorn Group . The lithology of the Hawthorn varies greatly between areas, but mostly consists of phosphatic clay, silt, and sand that range in color from cream or gray to green to brown ." (Williams and Kuniansky, 2016). "The entire formation forms a thick, generally clastic, highly variable sequence of lower permeability rock that, where present, is considered to be the upper confining unit of the Floridan aquifer system ." (Williams and Kuniansky, 2016).
Upper Floridan aquifer	Carbonate aquifer	"The Upper Floridan aquifer includes the uppermost or shallowest permeable zones in the Floridan aquifer system." (Williams, and Kuniansky, 2016) "The Floridan aquifer system includes the vertically continuous carbonate-rock system described by Miller (1986)". (Williams, and Kuniansky, 2016). "the carbonate rocks of the Upper Floridan aquifer may directly overlie clastic rocks of the Lisbon and Claiborne aquifers;" (Williams, and Kuniansky, 2016). "In Georgia, the Gulf Trough forms a narrow, distinct structural low at the top of the Floridan aquifer system (figs. 10 and 22). (Williams and Kuniansky, 2016).
Gordon confining unit	Sedimentary rock aquitard (consolidated or semi-consolidated rock)	"The Gordon aquifer is overlain and confined by the late-middle Eocene Lisbon-McBean confining unit and is equivalent to the Lisbon Formation in western and central Georgia, where it generally consists of massive glauconitic marl interbedded with calcareous clayey sand and fossiliferous limestone over most of its extent (Brooks and others, 1985)." (Williams and Kuniansky, 2016).
Lisbon-Avon Park composite unit	Sedimentary rock aquitard (consolidated or	"As shown in figure 6, the Upper and Lower Floridan aquifers are separated by the discontinuous numbered middle confining units that are now grouped within two composite units: (1) the Lisbon-Avon Park composite unit (LISAPCU) ,

Formation name	Category	Quote
	semi-consolidated rock	consisting of lower permeability clastic and higher to lower permeability carbonate rocks in the northern part of the study area; and (2) the middle Avon Park composite unit (MAPCU), consisting of lower permeability evaporite- and non-evaporite-bearing carbonate rocks to moderately permeable carbonate rocks in the central and southern parts of the study area." (Williams and Kuniansky, 2016).
Gordon aquifer	Sedimentary rock aquifer (consolidated or semi-consolidated rock)	"In Georgia, the Gordon aquifer consists of middle Eocene interbedded sand, silt, and clay and is equivalent to the Hatchetigbee and Tallahatta Formations and the lower part of the Lisbon Formation in western Georgia (Brooks and others, 1985)." (Williams and Kuniansky, 2016).
Wilcox confining zone	Sedimentary rock aquitard (consolidated or semi-consolidated rock)	"In Georgia, the base is identified as the Wilcox confining zone , which separates the overlying Lower Floridan and Claiborne aquifers from the underlying Clayton aquifer (Clarke and others, 1984)." (Williams and Kuniansky, 2016). "The Floridan is underlain everywhere by low-permeability rocks called the lower confining unit , which separates the Floridan aquifer system from older, deeper aquifers of the Southeastern Coastal Plain aquifer system." (Williams and Kuniansky, 2016). "The base of the Floridan aquifer system is marked by the lower confining unit , consisting of predominantly low-permeability late Paleocene to middle Eocene rocks ." (Williams and Kuniansky, 2016).
Light-gray very fine crystalline limestone, dark green glauconite	Sedimentary rock aquitard (consolidated or semi-consolidated rock)	" Glauconite Marker Unit: Peninsular Florida —Over much of peninsular Florida, a distinctive lower permeability unit lies deep within the Floridan aquifer system near its base ." (Williams and Kuniansky, 2016).
Paleocene rocks	Carbonate aquifer	" Paleocene rocks in the study area can be categorized into three groups. The first group, which is the most important to this study, primarily consists of interbedded dolomite and anhydrite of the Cedar Keys Formation , which underlies part of southeastern Georgia and all of peninsular Florida." (Williams and Kuniansky, 2016). " Interbedded dolomite and anhydrite, both being of lower permeability , compose the lower two-thirds of the Cedar Keys Formation of peninsular Florida and southeastern Georgia. This carbonate-evaporite sequence forms the base of the Floridan aquifer system ." (Williams and Kuniansky, 2016).
Cretaceous Formation (Navarro and Taylor rocks)	Sedimentary rock aquitard (consolidated or semi-consolidated rock)	"South of Brunswick, Ga., the base of the system drops several hundred feet into late Cretaceous rocks consisting of soft, friable, possibly Tayloran age, limestone with the permeable, late Cretaceous, Navarroan age Lawson Limestone above included as part of the Lower Floridan aquifer (Miller, 1986)." (Williams and Kuniansky, 2016). Figure 2 of (Williams and Kuniansky, 2016) under "Hydrogeology" in "Florida" " Confining unit ". (William, Raines, Lanning, 2016) under the "Hydrogeology" in "Florida" " confining unit ".

Williams, L.J., and Kuniansky, E.L. (2016). Revised hydrogeologic framework of the Floridan aquifer system in Florida and parts of Georgia, Alabama, and South Carolina (ver. 1.1, March 2016): U.S. Geological Survey Professional Paper 1807, 140 p., 23 pls., <http://dx.doi.org/10.3133/pp1807>

3.19 Catahoula Area, Gulf Coast Regional Aquifer System

Supplementary Fig. 135. Hydrogeologic cross section. 20 equally spaced transparent pink bars overlie the cross section; each shaded bar depicts the vertical offset from the land surface to the top of the uppermost confining unit or endogenous bedrock.

Supplementary Fig. 136. Vertical variations in the prevalence of wells that have been defined as tapping an unconfined or a confined aquifer by the USGS. The smaller squares represent 10 m depth intervals from the land surface to 100 m; the larger squares represent 20 m intervals from 100 m to 300 m below the land surface.

The Catahoula Area of the broader Gulf Coast Regional Aquifer System is located in southeastern Mississippi.

(i) A hydrogeologic cross section (based on Fig. 5 by Halford and Barber, 1995) suggests that the median depth to the uppermost

sedimentary aquitard is 40-50 meters below land surface (see pink transparent bars in cross section to the left).

(ii) We analysed wells within the study area that the USGS has defined as either unconfined or confined. The available USGS well data (n=3 wells) are insufficient to evaluate the depths at which the aquifer system transitions from unconfined to confined conditions.

(iii) In a regional-scale hydrogeologic study, Halford and Barber (1995) comment on the presence of confined conditions in the Catahoula formations; they state (quote): "The confining units overlying the upper, middle, and lower Catahoula aquifers are primarily clays within the Catahoula, Pascaoula, and Hattiesburg Formations."

Depth to confined conditions: 40-50 m (see (i) above)

Reference: Halford, K. J., Barber, N. L. (1995). U.S. Geological Survey Water-Resources Investigations Report 94-4219, 78 pp. Accessed March-2021 from <https://pubs.usgs.gov/wri/1994/4219/report.pdf>

The table below presents a series of published quotes (see quotation marks denoting text quoted from another publication, which is cited following the quotation marks with the full reference written in full below the table). The leftmost column lists a title of a hydrogeologic formation depicted in the cross section on the previous page. The rightmost column presents a quote from a hydrogeological study (see base of table for citation). The quote has been annotated with colored text to highlight how we categorized each layer (i.e., see categories in the center column in the table). Specifically: (i) blue text highlights portions of a quote that provide insights into the degree of consolidation of the formation, (ii) red text highlights portions of a quote that categorize the formation as an aquifer or an aquitard (i.e., higher versus lower permeability in the context of local hydrogeologic formations), and (iii) green text highlights portions of a quote that provide information about the lithology of the formation.

Supplementary Table 22. Hydrostratigraphy details for Catahoula Area

Formation name	Category	Quote
Land surface (surficial deposit)	Unconsolidated aquifer (In some area this aquifer is confined aquifer)	"Geologic units at land surface in the study area are sediments of Miocene age consisting of a complex series of alternating and lenticular beds of sand and clay, and other sediments of Pliocene and younger age." (Halford & Barber, 1995). "Thin surficial deposits of sand and gravel of the Citronelle Formation overlie the Pascagoula and Hattiesburg Formations in the immediate vicinity of the dome (fig. 23). Freshwater sands of the Pascagoula and Hattiesburg overlie the Catahoula Sandstone , the basal unit of the sedimentary beds that extend across the dome." (Spiers & Gandl, 1980).
Confining layer	Sedimentary rock aquitard (consolidated or semi-consolidated rock)	"The confining units overlying the upper, middle, and lower Catahoula aquifers are primarily clays within the Catahoula, Pascagoula, and Hattiesburg Formations. These clays vary in thickness from nearly zero to several hundred feet (figs. 9-11)." Halford & Barber, (1995).
Upper Catahoula aquifer	Clastic sedimentary rock aquifer (consolidated or semi-consolidated rock)	"The principal sources of ground water in the Laurel and Hattiesburg areas are sands within the Catahoula Formation which compose the Catahoula aquifer system. Generally the Catahoula Formation contains a thick water-bearing sand near the base of the formation and two other sands higher in the sequence; however the sands generally cannot be correlated regionally. Between and within the sands are units of clay that vary in thickness and areal extent. Following Boswell and others (1987, p. 18), the sands in this report are referred to as the lower, middle, and upper Catahoula aquifers. " Halford & Barber, (1995).
Confining layer	Sedimentary rock aquitard (consolidated or semi-consolidated rock)	"The confining units overlying the upper, middle, and lower Catahoula aquifers are primarily clays within the Catahoula, Pascagoula, and Hattiesburg Formations. These clays vary in thickness from nearly zero to several hundred feet (figs. 9-11)." Halford & Barber, (1995).
Middle Catahoula aquifer	Clastic sedimentary rock aquifer (consolidated or semi-consolidated rock)	"The principal sources of ground water in the Laurel and Hattiesburg areas are sands within the Catahoula Formation which compose the Catahoula aquifer system. Generally the Catahoula Formation contains a thick water-bearing sand near the base of the formation and two other sands higher in the sequence; however the sands generally cannot be correlated regionally. Between and within the sands are units of clay that vary in thickness and areal extent. Following Boswell and others (1987, p. 18), the sands in this report are

Formation name	Category	Quote
		referred to as the lower, middle, and upper Catahoula aquifers. " Halford & Barber, (1995).
Confining layer	Sedimentary rock aquitard (consolidated or semi-consolidated rock)	"The confining units overlying the upper, middle, and lower Catahoula aquifers are primarily clays within the Catahoula, Pascagoula, and Hattiesburg Formations. These clays vary in thickness from nearly zero to several hundred feet (figs. 9-11). " Halford & Barber, (1995).
Lower Catahoula aquifer	Clastic sedimentary rock aquifer (consolidated or semi-consolidated rock)	"The principal sources of ground water in the Laurel and Hattiesburg areas are sands within the Catahoula Formation which compose the Catahoula aquifer system. Generally the Catahoula Formation contains a thick water-bearing sand near the base of the formation and two other sands higher in the sequence; however the sands generally cannot be correlated regionally. Between and within the sands are units of clay that vary in thickness and areal extent. Following Boswell and others (1987, p. 18), the sands in this report are referred to as the lower, middle, and upper Catahoula aquifers. " Halford & Barber, (1995).
Base of Catahoula aquifer	Sedimentary rock aquitard (consolidated or semi-consolidated rock)	"The base of the lower Catahoula aquifer system in the study area was identified using borehole geophysical logs and the Glendon Formation, a highly resistive limestone unit within the Vicksburg Group, as the primary marker bed. " Halford & Barber, (1995).

Halford, K. J., Barber, N. L. (1995). *Analysis of ground-water flow in the Catahoula aquifer system in the vicinity of Laurel and Hattiesburg, Mississippi* (Vol. 94, No. 4219). US Department of the Interior, US Geological Survey. | Spiers, C. A., & Gandl, L. A. (1980). *A preliminary report of the geohydrology of the Mississippi salt-dome basin* (Vol. 80, No. 595). US Geological Survey. <https://pubs.er.usgs.gov/publication/ofr80595>

3.20 Houston-Galveston Area, Gulf Coast Regional Aquifer System

Supplementary Fig. 137. Hydrogeologic cross section. 20 equally spaced transparent pink bars overlie the cross section; each shaded bar depicts the vertical offset from the land surface to the top of the uppermost confining unit or endogenous bedrock.

Supplementary Fig. 138. Vertical variations in the prevalence of wells that have been defined as tapping an unconfined or a confined aquifer by the USGS. The smaller squares represent 10 m depth intervals from the land surface to 100 m; the larger squares represent 20 m intervals from 100 m to 300 m below the land surface.

The Houston-Galveston area is located near the central coast of Texas.

(i) A hydrogeologic cross section presented Fig. 2 by Braun et al. (2019) does not depict a clear confining unit within the aquifer system. However, the uppermost Chicot formation includes low-permeability units that together serve as confining units, leading to confined conditions in some wells screened within the Chicot Aquifer (see (ii) below).

(ii) We analysed wells within the study area that the USGS has defined as either unconfined or confined. Most (>80%) wells at depths of 20-30 m and at depths exceeding 20 m are defined as tapping a confined aquifer.

Depth to confined conditions: 20-30 m (see (ii) above)

Reference: Braun, C.L., Ramage, J.K., Shah, S.D., (2019). Status of groundwater-level altitudes and long-term groundwater level changes in the Chicot, Evangeline, and Jasper aquifers, Houston-Galveston region, Texas, 2019. US Geological Survey Scientific Investigations Report 2019-5089, 18 pp., <https://doi.org/10.3133/sir20195089>

The table below presents a series of published quotes (see quotation marks denoting text quoted from another publication, which is cited following the quotation marks with the full reference written in full below the table). The leftmost column lists a title of a hydrogeologic formation depicted in the cross section on the previous page. The rightmost column presents a quote from a hydrogeological study (see base of table for citation). The quote has been annotated with colored text to highlight how we categorized each layer (i.e., see categories in the center column in the table). Specifically: (i) blue text highlights portions of a quote that provide insights into the degree of consolidation of the formation, (ii) red text highlights portions of a quote that categorize the formation as an aquifer or an aquitard (i.e., higher versus lower permeability in the context of local hydrogeologic formations), and (iii) green text highlights portions of a quote that provide information about the lithology of the formation.

Supplementary Table 23. Hydrostratigraphy details for the Houston Galveston Area

Formation name	Category	Quote
Chicot aquifer	Sedimentary rock aquifer (consolidated or semi-consolidated rock)	"It consists of several aquifers , including the Jasper, Evangeline, and Chicot aquifers , which are composed of discontinuous sand, silt, clay, and gravel beds of Miocene to Holocene age (Figure 6-17)." (Bruun et al., 2017).
Evangeline aquifer	Sedimentary rock aquifer (consolidated or semi-consolidated rock)	"It consists of several aquifers , including the Jasper, Evangeline , and Chicot aquifers , which are composed of discontinuous sand, silt, clay, and gravel beds of Miocene to Holocene age (Figure 6-17)." (Bruun et al., 2017).
Burkeville confining unit	Sedimentary rock aquitard (consolidated or semi-consolidated rock)	"The Oligocene Catahoula tuff forms a leaky confining layer at the base of the aquifer, and the Burkeville confining unit separates the Jasper Aquifer from the Evangeline Aquifer. All of the sedimentary units thicken toward the Gulf of Mexico." (Bruun et al., 2017). "The updated Burkeville Confining Unit consists primarily of the clay-rich deposits comprising the Upper, Middle, and Lower Lagarto formations delineated by Young and others (2010, 2012)." (Young, 2020)
Jasper aquifer	Sedimentary rock aquifer (consolidated or semi-consolidated rock)	"It consists of several aquifers , including the Jasper , Evangeline, and Chicot aquifers , which are composed of discontinuous sand, silt, clay, and gravel beds of Miocene to Holocene age (Figure 6-17)." (Bruun et al., 2017).
Catahoula confining system	Sedimentary rock aquitard (consolidated or semi-consolidated rock)	"The Oligocene Catahoula tuff forms a leaky confining layer at the base of the aquifer, and the Burkeville confining unit separates the Jasper Aquifer from the Evangeline Aquifer. All of the sedimentary units thicken toward the Gulf of Mexico." (Bruun et al., 2017). "one for each of the hydrogeologic units of the aquifer system except the Catahoula confining system, the assumed no-flow base of the system. " Kasmarek (2021)

Bruun, B., Jackson, K., Lake, P., Walker, J. (2016). Texas Aquifers Study. Texas Water Development Board Report. 336 pp. Accessed June 14, 2022 from https://www.twdb.texas.gov/groundwater/docs/studies/TexasAquifersStudy_2016.pdf#page=89

Grubb, H. F. (1984). Planning report for the Gulf Coast Regional Aquifer-System Analysis in the Gulf of Mexico Coastal Plain, United States. Water-Resources Investigations Report, 84, 4219. Accessed June 14, 2022 from <https://pubs.usgs.gov/wri/1984/4219/report.pdf>

[Young C.S. \(2020\). The delineation of the Burkeville confining unit and the base of the Chicot aquifer to support the development of the Gulf 2023 groundwater model. INTERA Incorporated. Accessed June 14, 2022 from https://hgsubsidence.org/wp-content/uploads/2021/06/Final_HGSD_FBSD_Burkeville_Report_final.pdf](https://hgsubsidence.org/wp-content/uploads/2021/06/Final_HGSD_FBSD_Burkeville_Report_final.pdf)

[Kasmarek, M.C. \(2012\). Hydrogeology and simulation of groundwater flow and land-surface subsidence in the northern part of the Gulf Coast aquifer system, Texas, 1891–2009 \(ver. 1.1, November 2013\): U.S. Geological Survey Scientific Investigations Report 2012–5154, 55 p., http://pubs.usgs.gov/sir/2012/5154/. Accessed June 14, 2022 from https://www.twdb.texas.gov/groundwater/models/gam/qlfc_n/HAGM.SIR.Version1.1.November2013.pdf](http://pubs.usgs.gov/sir/2012/5154/)

[Braun, C.L., Ramage, J.K., Shah, S.D. \(2019\). Status of groundwater-level altitudes and long-term groundwaterlevel changes in the Chicot, Evangeline, and Jasper aquifers, Houston-Galveston region, Texas, 2019: U.S. Geological Survey Scientific Investigations Report 2019–5089, 18 p., https://doi.org/10.3133/sir20195089](https://doi.org/10.3133/sir20195089)

3.21 Lafayette Area, Gulf Coast Regional Aquifer System

Supplementary Fig. 139. Hydrogeologic cross section. 20 equally spaced transparent pink bars overlie the cross section; each shaded bar depicts the vertical offset from the land surface to the top of the uppermost confining unit or endogenous bedrock.

Supplementary Fig. 140. Vertical variations in the prevalence of wells that have been defined as tapping a confined or a confined aquifer by the USGS. The smaller squares represent 10 m depth intervals from the land surface to 100 m; the larger squares represent 20 m intervals from 100 m to 300 m below the land surface.

The Lafayette Area is located in southwestern Louisiana.

(i) A hydrogeologic cross section presented in Fig. 7 by Vahdat-Aboueshagh et al. (2021) suggests that a low permeability clay-rich unit exists near to the land surface in the study area.

(ii) We analysed wells within the study area that the USGS has defined as either unconfined or confined. Most (>80%) wells at depths of 20-30 m and at depths exceeding 20 m are defined as tapping a confined aquifer.

Depth to confined conditions: 20-30 m (see (ii) above)

Reference: Vahdat-Aboueshagh, H., Tsai, F. T. C., Bhatta, D., Paudel, K. P. (2021). Irrigation-intensive groundwater modeling of complex aquifer systems through integration of big geological data. *Frontiers in Water*, 3, 623476.

The table below presents a series of published quotes (see quotation marks denoting text quoted from another publication, which is cited following the quotation marks with the full reference written in full below the table). The leftmost column lists a title of a hydrogeologic formation depicted in the cross section on the previous page. The rightmost column presents a quote from a hydrogeological study (see base of table for citation). The quote has been annotated with colored text to highlight how we categorized each layer (i.e., see categories in the center column in the table). Specifically: (i) blue text highlights portions of a quote that provide insights into the degree of consolidation of the formation, (ii) red text highlights portions of a quote that categorize the formation as an aquifer or an aquitard (i.e., higher versus lower permeability in the context of local hydrogeologic formations), and (iii) green text highlights portions of a quote that provide information about the lithology of the formation.

Supplementary Table 24. Hydrostratigraphy details for the Lafayette Area

Formation name	Category	Quote
Confining clay units	Sedimentary rock aquitard (consolidated or semi-consolidated rock)	"The Pleistocene Upper Chicot and Lower Chicot aquifers are separated by clay beds as thick as 30m (100 ft) in a few areas . The clay beds are not extensive (Williams and Duex, 1995). In general, sands in the north are relatively shallow and often interbedded with clays ." (Vahdat-Aboueshagh et al., 2021).
Upper Chicot aquifer	Unconsolidated aquifer	"The Chicot aquifer system encompasses seven aquifers (Nyman et al., 1990): the Upper Chicot and the Lower Chicot aquifers in the east, the Undifferentiated sand in the central area, and the Shallow sand, the "200-foot" sand, the "500-foot" sand, and the "700-foot" sand in the west." (Vahdat-Aboueshagh et al., 2021). "The Chicot is comprised of a series of unconsolidated sands, gravels, silts and clays of Holocene through Pliocene-age that were generated by the ancestral Mississippi River to the east and the smaller Sabine and Calcasieu Rivers to the west (Prakken, 2003, Tollett et al., 2003, Lovelace et al., 2004; Fig. 1)." (Borrok et al., 2016).
Lower Chicot aquifer	Unconsolidated aquifer	"The Chicot aquifer system encompasses seven aquifers (Nyman et al., 1990): the Upper Chicot and the Lower Chicot aquifers in the east, the Undifferentiated sand in the central area, and the Shallow sand, the "200-foot" sand, the "500-foot" sand, and the "700-foot" sand in the west." (Vahdat-Aboueshagh et al., 2021). "The Chicot is comprised of a series of unconsolidated sands, gravels, silts and clays of Holocene through Pliocene-age that were generated by the ancestral Mississippi River to the east and the smaller Sabine and Calcasieu Rivers to the west (Prakken, 2003, Tollett et al., 2003, Lovelace et al., 2004; Fig. 1)." (Borrok et al., 2016).
Sedimentary aquifer (maybe the Evangeline aquifer)	Sedimentary rock aquifer (consolidated or semi-consolidated rock)	"Pumping at a rate that is greater than can be supported by the Chicot in this region likely resulted in the capture of Na-HCO ₃ -rich waters from the underlying Evangeline aquifer ." (Borrok et al., 2016). "The Chicot, Evangeline , and Jasper equivalent aquifer systems extend across most of southeastern

Formation name	Category	Quote
		Louisiana and generally consist of silt, sand, and gravel separated by discontinuous layers of clay and sandy clay. " USGS (2017).

Vahdat-Aboueshagh, H., Tsai, F. T. C., Bhatta, D., Paudel, K. P. (2021). Irrigation-Intensive groundwater modeling of complex aquifer systems through integration of big geological data. *Frontiers in Water*, 3, 623476.

Borrok, D. M., Broussard III, W. P. (2016). Long-term geochemical evaluation of the coastal Chicot aquifer system, Louisiana, USA. *Journal of Hydrology*, 533, 320-331.

USGS (2017). Water Resources of the Southern Hills Regional Aquifer System, Southeastern Louisiana. US Geological Survey Fact Sheet Fact Sheet 2017-3010, 6 pp. Accessed April 12, 2022 via <https://pubs.usgs.gov/fs/2017/3010/fs20173010.pdf>

3.22 Southern Hills, Gulf Coast Regional Aquifer System

Supplementary Fig. 141. Hydrogeologic cross section. 20 equally spaced transparent pink bars overlie the cross section; each shaded bar depicts the vertical offset from the land surface to the top of the uppermost confining unit or endogenous bedrock.

Supplementary Fig. 142. Vertical variations in the prevalence of wells that have been defined as tapping an unconfined or a confined aquifer by the USGS. The smaller squares represent 10 m depth intervals from the land surface to 100 m; the larger squares represent 20 m intervals from 100 m to 300 m below the land surface.

The Southern Hills subarea of the broader Gulf Coast Regional Aquifer System is located in southeastern Louisiana.

(i) A hydrogeologic cross section presented in Fig. 4 by USGS (2017) depicts a series of interbedded aquifers and low-permeability units.

(ii) We analysed wells within the study area that the USGS has defined as either unconfined or confined. Most (>80%) wells at depths of 20-30 m and at depths exceeding 20 m are defined as tapping a confined aquifer.

Depth to confined conditions: 20-30 m (see (ii) above)

Reference: USGS (2017). [Water Resources of the Southern Hills Regional Aquifer System, Southeastern Louisiana. US Geological Survey Fact Sheet Fact Sheet 2017-3010, 6 pp. Accessed April 12, 2022 via https://pubs.usgs.gov/fs/2017/3010/fs20173010.pdf](https://pubs.usgs.gov/fs/2017/3010/fs20173010.pdf)

The table below presents a series of published quotes (see quotation marks denoting text quoted from another publication, which is cited following the quotation marks with the full reference written in full below the table). The leftmost column lists a title of a hydrogeologic formation depicted in the cross section on the previous page. The rightmost column presents a quote from a hydrogeological study (see base of table for citation). The quote has been annotated with colored text to highlight how we categorized each layer (i.e., see categories in the center column in the table). Specifically: (i) blue text highlights portions of a quote that provide insights into the degree of consolidation of the formation, (ii) red text highlights portions of a quote that categorize the formation as an aquifer or an aquitard (i.e., higher versus lower permeability in the context of local hydrogeologic formations), and (iii) green text highlights portions of a quote that provide information about the lithology of the formation.

Supplementary Table 25. Hydrostratigraphy details for the Southern Hills aquifer

Formation name	Category	Quote
Chicot equivalent aquifer system	Unconsolidated aquifer	"The Chicot , Evangeline , and Jasper equivalent aquifer systems extend across most of southeastern Louisiana and generally consist of silt, sand, and gravel separated by discontinuous layers of clay and sandy clay ." USGS (2017). "" USGS (2017). " The primary aquifers composing the Chicot equivalent aquifer system in the western part of the 10-parish area, from shallowest to deepest, are the shallow sands, Upland terrace aquifer, and the "400-foot" and "600-foot" sands of the Baton Rouge area (Griffith, 2003; fig. 4)." USGS (2017).
Evangeline equivalent aquifer system	Sedimentary rock aquifer (consolidated or semi-consolidated rock)	"The Chicot , Evangeline , and Jasper equivalent aquifer systems extend across most of southeastern Louisiana and generally consist of silt, sand, and gravel separated by discontinuous layers of clay and sandy clay ." USGS (2017). " The primary aquifers composing the Evangeline equivalent aquifer system , from shallowest to deepest, in the western part of the 10-parish area are the "800-foot," "1,000-foot," "1,200-foot," "1,500-foot," and "1,700-foot" sands of the Baton Rouge area (Griffith, 2003; fig. 4)." USGS (2017).
Jasper equivalent aquifer system	Sedimentary rock aquifer (consolidated or semi-consolidated rock)	"The Chicot , Evangeline , and Jasper equivalent aquifer systems extend across most of southeastern Louisiana and generally consist of silt, sand, and gravel separated by discontinuous layers of clay and sandy clay ." USGS (2017). "" USGS (2017). " The primary aquifers composing the Jasper equivalent aquifer system , from shallowest to deepest, in the western part of the 10-parish area are the "2,000-foot," "2,400-foot," and "2,800-foot" sands of the Baton Rouge area (Griffith, 2003; fig. 4)." USGS (2017).

USGS (2017). Water Resources of the Southern Hills Regional Aquifer System, Southeastern Louisiana. US Geological Survey Fact Sheet Fact Sheet 2017-3010, 6 pp. Accessed April 12, 2022 via <https://pubs.usgs.gov/fs/2017/3010/fs20173010.pdf>

3.23 Central High Plains, High Plains

Supplementary Fig. 143. Hydrogeologic cross section. 20 equally spaced transparent pink bars overlie the cross section; each shaded bar depicts the vertical offset from the land surface to the top of the uppermost confining unit or endogenous bedrock.

Supplementary Fig. 144. Vertical variations in the prevalence of wells that have been defined as tapping an unconfined or a confined aquifer by the USGS. The smaller squares represent 10 m depth intervals from the land surface to 100 m; the larger squares represent 20 m intervals from 100 m to 300 m below the land surface.

The Central High Plains aquifer system is located in northwest Texas, southwest Kansas, western Oklahoma, northwest New Mexico, and southeast Colorado.

(i) A hydrogeologic cross section presented by Macfarlane (1996) suggests that the uppermost confining unit is 43 meters below the land surface (25th-75th percentile range: 0-97 meters below land surface).

(ii) We analysed wells within the study area that the USGS has defined as either unconfined or confined. Nearly all wells with depths of less than 300 m are defined as tapping unconfined aquifers. All wells (n=3) at depths of 300-340 m are defined as tapping a confined aquifer (Dakota Sandstone aquifer).

Depth to confined conditions: 300-340 m (see (ii) above)

References: Macfarlane, P. A. (1996). An analysis of the upper part of the regional flow system along the southern groundwater flow "corridor" in the Dakota aquifer using a steady-state, vertical profile flow model. http://www.kgs.ku.edu/Hydro/Publications/1996/OFR96_1d/index.html

The table below presents a series of published quotes (see quotation marks denoting text quoted from another publication, which is cited following the quotation marks with the full reference written in full below the table). The leftmost column lists a title of a hydrogeologic formation depicted in the cross section on the previous page. The rightmost column presents a quote from a hydrogeological study (see base of table for citation). The quote has been annotated with colored text to highlight how we categorized each layer (i.e., see categories in the center column in the table). Specifically: (i) blue text highlights portions of a quote that provide insights into the degree of consolidation of the formation, (ii) red text highlights portions of a quote that categorize the formation as an aquifer or an aquitard (i.e., higher versus lower permeability in the context of local hydrogeologic formations), and (iii) green text highlights portions of a quote that provide information about the lithology of the formation.

Supplementary Table 26. Hydrostratigraphy details for the Central High Plains

Formation name	Category	Quote
Alluvial Valley & High Plains aquifer	Unconsolidated aquifer	“Unconsolidated alluvial and eolian deposits” Macfarlane, P. A. (1996) “Alluvial Valley & High Plains aquifer ” Macfarlane, P. A. (1996)
Upper Cretaceous aquitard	Sedimentary rock aquitard (consolidated or semi-consolidated rock)	“Pierre Shale, Niobrara Chalk, Carlile Shale, Greenhorn Limestone, Graneros Shale” Macfarlane, P. A. (1996) “Upper Cretaceous aquitard” P. A. (1996)
Upper Dakota aquifer	Clastic sedimentary rock aquifer (consolidated or semi-consolidated rock)	“Buff to light-brown, fine to medium grained sandstone with interbedded shale. ” Luckey and Mark, 1999 “Upper Dakota aquifer” Macfarlane, P. A. (1996)
Kiowa shale aquitard	Sedimentary rock aquitard (consolidated or semi-consolidated rock)	“Gray to black shale with some fine-grained sandstone in upper part.” Luckey and Mark, 1999. “Kiowa shale aquitard” Macfarlane, P. A. (1996)
Lower Dakota aquifer	Clastic sedimentary rock aquifer (consolidated or semi-consolidated rock)	“Longford member” Macfarlane, P. A. (1996) “Cheyenne Sandstone” Macfarlane, P. A. (1996) “White to buff, fine to medium-grained sandstone with some interbedded shales. Unit contains some conglomerate in lower part.” Luckey and Mark, 1999. “Lower Dakota aquifer” Macfarlane, P. A. (1996)
Morrison-Dockum aquifer	Clastic sedimentary rock aquifer (consolidated or semi-consolidated rock)	“Morrison Formation Luckey and Mark, 1999. “Varicolored shale, sandstone, limestone, dolostone, and conglomerate.” Luckey and Mark, 1999. “The Dockum Group is composed of sandstone with interbedded shales grading upward to a shaly sandstone or siltstone Luckey and Mark, 1999. “The Dakota Sandstone, the Lytle Sandstone, and the Dockum Group all provide sufficient water for stock and domestic use and may provide sufficient water for irrigation, particularly when combined with the High Plains aquifer or with each other. ” Luckey and Mark, 1999. “Morrison-Dockum aquifer” Macfarlane, P. A. (1996)
Permian Aquitard	Sedimentary rock aquitard (consolidated or semi-consolidated rock)	“Undifferentiated Permian” Macfarlane, P. A. (1996) “Undifferentiated red beds” Luckey et al., 1999. “Predominately red or orange, shale, mudstone, siltstone, sandstone, dolostone, and anhydrite

Formation name	Category	Quote
		with some gypsum, limestone and halite." Luckey et al., 1999. "Permian Aquitard" Macfarlane, P. A. (1996)
Cedar Hills Sandstone	Clastic sedimentary rock aquifer (consolidated or semi-consolidated rock)	"Cedar Hill Sandstone" Macfarlane, P. A. (1996) "Cedar Hills Sandstone aquifer" Macfarlane, P. A. (1996)

Luckey, R.L., Becker, M.F. (1999). Hydrogeology, water use, and simulation of flow in the High Plains aquifer in northwestern Oklahoma, southeastern Colorado, southwestern Kansas, northeastern New Mexico, and northwestern Texas. US Geological Survey Water-Resources Investigations Report 99-4104, 73 pp. Accessed February 21, 2022 from <https://pubs.usgs.gov/wri/wri994104/pdf/wri994104.pdf>

Macfarlane, P.A. (1996). An analysis of the upper part of the regional flow system along the southern ground-water flow "corridor" in the Dakota aquifer using a steady-state, vertical profile flow model. Open-File Report. http://www.kgs.ku.edu/Hydro/Publications/1996/OFR96_1d/OFR96-1d.pdf

3.24 Northern High Plains, High Plains

Supplementary Fig. 145. Hydrogeologic cross section. 20 equally spaced transparent pink bars overlie the cross section; each shaded bar depicts the vertical offset from the land surface to the top of the uppermost confining unit or endogenous bedrock.

Supplementary Fig. 146. Vertical variations in the prevalence of wells that have been defined as tapping an unconfined or a confined aquifer by the USGS. The smaller squares represent 10 m depth intervals from the land surface to 100 m; the larger squares represent 20 m intervals from 100 m to 300 m below the land surface.

The Northern High Plains aquifer system is located, predominantly, in Nebraska.

(i) A hydrogeologic cross section presented in Fig. 2 by Hallum et al. (2019) depicts relatively deep depths (>200 m) to the uppermost relatively low-permeability unit.

(ii) We analysed wells within the study area that the USGS has defined as either unconfined or confined. Nearly all wells with depths of less than 240 m are defined as tapping unconfined aquifers. One well with a depth of 300 m is defined as tapping a confined aquifer (Dakota Sandstone aquifer).

Depth to confined conditions: >240 m (see (ii) above)

Reference: Hallum, D.R., Terry D.O., Leite, M.B., Sibray, S.S., Balmat, J.L. (2019) Exploring the northwest margin of the High Plains Aquifer: Structure, depositional history and hydrogeology. Nebraska Geological Society Field Trip, University of Nebraska–Lincoln, Conservation and Survey Division, 43 pp.

The table below presents a series of published quotes (see quotation marks denoting text quoted from another publication, which is cited following the quotation marks with the full reference written in full below the table). The leftmost column lists a title of a hydrogeologic formation depicted in the cross section on the previous page. The rightmost column presents a quote from a hydrogeological study (see base of table for citation). The quote has been annotated with colored text to highlight how we categorized each layer (i.e., see categories in the center column in the table). Specifically: (i) blue text highlights portions of a quote that provide insights into the degree of consolidation of the formation, (ii) red text highlights portions of a quote that categorize the formation as an aquifer or an aquitard (i.e., higher versus lower permeability in the context of local hydrogeologic formations), and (iii) green text highlights portions of a quote that provide information about the lithology of the formation.

Supplementary Table 27. Hydrostratigraphy details for the Northern High Plains

Formation name	Category	Quote
Quaternary-age deposit	Unconsolidated aquifer	"Quaternary-age deposits include alluvial, valley-fill, dune sand, and glacial deposits (fig. 6)." (Peterson et al., 2020) Valley-fill deposits: " Stream-laid deposits of gravel, sand, silt, and clay associated with the most recent cycle of erosion and deposition along present streams . Usually hydraulically connected to underlying Quaternary and Tertiary deposits." (Peterson et al., 2020). Dune sand: " Fine to medium sand with small amounts of clay, silt, and coarse sand formed into hills and ridges by the wind ." (Peterson et al., 2020) Loess: " Silt with lesser amounts of very fine sand and clay deposited as windblown dust . Where thickest, generally only the lowest 100 feet is below the water table." (Peterson et al., 2020). "Unconsolidated alluvial deposit: Stream-laid deposits of gravel, sand, silt, and clay locally cemented by calcium carbonate into caliche or mortar beds ." (Peterson et al., 2020). " Usually hydraulically connected to underlying Quaternary and Tertiary deposits ." (Peterson et al., 2020)
Ogallala Formation	Unconsolidated aquifer	" Poorly sorted clay, silt, sand, and gravel generally unconsolidated ; forms caliche layers or mortar beds when cemented by calcium carbonate. Most of the saturated thickness of the Northern High Plains aquifer, though absent from the western and eastern ends." (Peterson et al., 2020) " The Ogallala Formation composes most of the Northern High Plains aquifer (fig. 6, table 2; Gutentag and others, 1984) ." (Peterson et al., 2020)
Arikaree Group	Clastic sedimentary rock aquifer (consolidated or semi-consolidated rock)	"Predominantly massive very fine to fine-grained sandstone with localized beds of volcanic ash, silty sand, siltstone, claystone, sandy clay, limestone, marl, and mortar beds ." (Peterson et al., 2020) " Ogallala and Arikaree Group strata that comprise the High Plains Aquifer ." (Hallum et al., 2019).
White River Group	Sedimentary rock aquitard (consolidated or semi-consolidated rock) (may be aquifer where secondary permeability resulting	"Upper unit, Brule Formation, predominantly massive siltstone containing sandstone beds and channel deposits of sandstone . Included where it contains saturated sandstones or interconnected fractures, mainly limited to western Nebraska. Otherwise, the top of the Brule Formation is considered the base of the Northern High Plains aquifer." (Peterson et al., 2020) " The Chadron and Brule Formations of the White River Group, together with the younger Arikaree Group, are generally fine-grained, low-

Formation name	Category	Quote
	interconnected fractures - (Peterson et al., 2020, Gutentag et al., 1984).	permeability units except for a few areas of high permeability and areas where permeability has been increased by fractures. " (Peterson et al., 2020)
Cretaceous rock (undifferentiated)	Sedimentary rock aquitard (consolidated or semi-consolidated rock)	"Shales, chinks, limestones, or other poorly permeable deposits, the top of which form the base of the Northern High Plains aquifer. Most often, Pierre Shale, Niobrara Chalk, or Niobrara Shale. " (Peterson et al., 2020)
Maha (Dakota) aquifer	Clastic sedimentary rock aquifer (consolidated or semi-consolidated rock)	"The upper aquifer of the Great Plains aquifer system is called the Maha aquifer. This aquifer was formerly called the Dakota aquifer from the Dakota Sandstone, which is a prominent part of the aquifer." (Miller & Appel, 1997). "The Maha and the Apishapa aquifers consist of loosely cemented, medium- to fine-grained sandstone." (Miller & Appel, 1997). "The Maha aquifer is more extensive than the Apishapa aquifer (fig. 76);" (Miller & Appel, 1997).
Great Plains aquifer system	Clastic sedimentary rock aquifer (consolidated or semi-consolidated rock)	"The Great Plains aquifer system is exposed at the land surface in a band that extends from south-central Kansas to northeastern Nebraska (fig. 5). This aquifer system consists of two sandstone aquifers in Cretaceous rocks, separated by a shale confining unit. " (Miller & Appel, 1997). "Water in the Great Plains aquifer system is under confined conditions in most places." (Miller & Appel, 1997). "Sand bodies are typically linear, lenticular, or sinuous, which indicates that they were deposited in deltaic, shoreline, or fluvial environments."

Hallum, D.R., Terry D.O., Leite, M.B., Sibray, S.S., Balmat, J.L. (2019). Exploring the northwest margin of the High Plains Aquifer: Structure, depositional history and hydrogeology. Nebraska Geological Society Field Trip, University of Nebraska–Lincoln, Conservation and Survey Division, 43 pp.

Peterson, S.M., Traylor, J.P., Guira, M. (2020). Groundwater Availability of the Northern High Plains Aquifer in Colorado, Kansas, Nebraska, South Dakota, and Wyoming. U.S. Geological Survey Professional Paper 1864, 57 p., Accessed November 28, 2021 from <https://pubs.usgs.gov/pp/1864/pp1864.pdf>

Gutentag, E.D., Heimes, F.J., Krothe, N. C., Luckey, R. R., Weeks, J. B. (1984). Geohydrology of the High Plains aquifer in parts of Colorado, Kansas, Nebraska, New Mexico, Oklahoma, South Dakota, Texas, and Wyoming (No. 1400-B). Accessed February 10, 2021 from <https://pubs.usgs.gov/pp/1400b/report.pdf>

Miller, J.A., Appel, C.L. (1997). *Ground water atlas of the United States: Segment 3, Kansas, Missouri, Nebraska* (No. 730-D, pp. D1-D24). US Geological Survey. Accessed April 22, 2022 from <https://pubs.er.usgs.gov/publication/ha730D>

3.25 Southern High Plains, High Plains

Supplementary Fig. 147. Hydrogeologic cross section. 20 equally spaced transparent pink bars overlie the cross section; each shaded bar depicts the vertical offset from the land surface to the top of the uppermost confining unit or endogenous bedrock.

Supplementary Fig. 148. Vertical variations in the prevalence of wells that have been defined as tapping an unconfined or a confined aquifer by the USGS. The smaller squares represent 10 m depth intervals from the land surface to 100 m; the larger squares represent 20 m intervals from 100 m to 300 m below the land surface.

The Southern High Plains aquifer system is located in eastern New Mexico and western Texas.

(i) A hydrogeologic cross section presented in Fig. 15 by Blandford et al. (2008) suggests that the aquifer system does not depict a clear confining unit within the shallow (<200 m) portion of the aquifer system.

(ii) We analysed wells within the study area that the USGS has defined as either unconfined or confined. Nearly all wells are defined as tapping an unconfined aquifer, including the deepest well in the dataset (274 m).

Depth to confined conditions: >270 m (see (ii) above)

Reference: Blandford, T. N., Kuchanur, M., Standen, A., Ruggiero, R., Calhoun, K. C., Kirby, P., Shah, G. (2008). Groundwater Availability Model of the Edwards-Trinity (High Plains) Aquifer in Texas and New Mexico. Texas Water Development Board. <https://www.twdb.texas.gov/groundwater/aquifer/minors/edwards-trinity-high-plains.asp>

The table below presents a series of published quotes (see quotation marks denoting text quoted from another publication, which is cited following the quotation marks with the full reference written in full below the table). The leftmost column lists a title of a hydrogeologic formation depicted in the cross section on the previous page. The rightmost column presents a quote from a hydrogeological study (see base of table for citation). The quote has been annotated with colored text to highlight how we categorized each layer (i.e., see categories in the center column in the table). Specifically: (i) blue text highlights portions of a quote that provide insights into the degree of consolidation of the formation, (ii) red text highlights portions of a quote that categorize the formation as an aquifer or an aquitard (i.e., higher versus lower permeability in the context of local hydrogeologic formations), and (iii) green text highlights portions of a quote that provide information about the lithology of the formation.

Supplementary Table 28. Hydrostratigraphy details for the Southern High Plains

Formation name	Category	Quote
Tertiary Ogallala Formation	Unconsolidated aquifer	" Tan, yellow, and reddish brown silt, clay, sand, and gravel. Caliche layers common near the surface." (Table 1, Blandford, 2008). " Yields moderate to large amounts of water to wells across Southern High Plains. " (Table 1, Blandford, 2008). "Depositional environments of the Ogallala Formation have been interpreted as including coalescing alluvial fans or alluvial aprons (Johnson, 1901; Frye and Leonard, 1964; Seni, 1980; Reeves, 1984) or fluvial-dominated valley fill sequences confined within paleovalleys (Gustavson, 1996)." (Blandford et al., 2008)
Cretaceous (Duck Creek and Kiamichi Formation are aquitard; Carbonate aquifer exists in Edwards and Comanche Peak Formations)	Clastic sedimentary rock aquifer (consolidated or semi-consolidated rock)	Edwards Formation: " Light gray to yellowish gray, thick to massive bedded, fine to coarse-grained limestone. " (Table 1, Blandford et al., 2008). Comanche Peak Formation: "Light gray to yellowish brown, irregularly bedded argillaceous limestone with thin interbeds of light gray shale. " For both Edwards and Comanche Peak Formation: "Generally yields fairly small amounts of water to wells beneath Southern High Plains, but may yield large amounts of water locally due to fractures and solution cavities. " (Table 1, Blandford et al., 2008). Cretaceous Trinity Group, Antlers Formation: " White, gray, yellowish brown to purple, argillaceous, loosely cemented sand, sandstone, and conglomerate with interbeds of siltstone and clay. " (Table 1, Blandford et al., 2008). " Yields small to moderate amounts of water to wells. " (Table 1, Blandford et al., 2008). Cretaceous System, Hydrogeologic Units: " Edwards-Trinity (High Plains) Aquifer " (Table 1, Blandford et al., 2008).
Triassic Dockum Group	Clastic sedimentary rock aquifer (consolidated or semi-consolidated rock)	"The Triassic section can be as much as 2,000 feet thick, and its low-permeability sediments in the upper portion of the section separate groundwater in the Southern Ogallala and Edwards- Trinity (High Plains) Aquifers from groundwater in Triassic sandstone units, referred to as the Dockum Aquifer. " (Blandford et al., 2008). " Multi-colored fine- to coarse grained micaceous sandstone with some claystone and shale interbeds. " (Table 1, Blandford et al., 2008).

Formation name	Category	Quote
Permian (evaporite beds exists)	Sedimentary rock aquitard (consolidated or semi-consolidated rock)	"buried Permian salt beds" (Blandford et al., 2008). "Bedrock units in contact with the High Plains aquifer range in age from Permian to Tertiary." (Gutentag et al., 1984).

Blandford, T. N., Kuchanur, M., Standen, A., Ruggiero, R., Calhoun, K. C., Kirby, P., Shah, G. (2008, December). Groundwater Availability Model of the Edwards-Trinity (High Plains) Aquifer in Texas and New Mexico. Texas Water Development Board. Accessed June 6, 2022 via <https://www.twdb.texas.gov/groundwater/aquifer/minors/edwards-trinity-high-plains.asp>

Gutentag, E.D., Heimes, F.J., Krothe, N.C., Luckey, R.R., Weeks, J.B. (1984). Geohydrology of the High Plains aquifer in parts of Colorado, Kansas, Nebraska, New Mexico, Oklahoma, South Dakota, Texas, and Wyoming (No. 1400-B). Accessed June 6, 2022 via <https://pubs.er.usgs.gov/publication/pp1400B>

3.26 Albuquerque Basin, Middle Rio Grande

Supplementary Fig. 149. Hydrogeologic cross section. 20 equally spaced transparent pink bars overlie the cross section; each shaded bar depicts the vertical offset from the land surface to the top of the uppermost confining unit or endogenous bedrock.

Supplementary Fig. 150. Vertical variations in the prevalence of wells that have been defined as tapping an unconfined or a confined aquifer by the USGS. The smaller squares represent 10 m depth intervals from the land surface to 100 m; the larger squares represent 20 m intervals from 100 m to 300 m below the land surface.

The Albuquerque Basin is located in central New Mexico.

(i) A hydrogeologic cross section presented in Plate 2 by Connell (2006) depicts thick (>200 m) poorly consolidated aquifers with a high density of faulting.

(ii) We analysed wells within the study area that the USGS has defined as either unconfined or confined. Nearly all wells in the study area are defined in the USGS dataset as tapping an unconfined aquifer, including the deepest well in the dataset (631 m).

Depth to confined conditions: >630 m (see (ii) above)

Reference: Connell, S. (2006). Preliminary Geologic Map of the Albuquerque-Rio Rancho Metropolitan Area & Vicinity, Bernalillo & Sandoval Counties, New Mexico. New Mexico Bureau of Geology and Mineral Resources Open-File Report 496. Accessed July 7, 2022 via <https://geoinfo.nmt.edu/publications/openfile/details.cfm?Volume=496>

The table below presents a series of published quotes (see quotation marks denoting text quoted from another publication, which is cited following the quotation marks with the full reference written in full below the table). The leftmost column lists a title of a hydrogeologic formation depicted in the cross section on the previous page. The rightmost column presents a quote from a hydrogeological study (see base of table for citation). The quote has been annotated with colored text to highlight how we categorized each layer (i.e., see categories in the center column in the table). Specifically: (i) blue text highlights portions of a quote that provide insights into the degree of consolidation of the formation, (ii) red text highlights portions of a quote that categorize the formation as an aquifer or an aquitard (i.e., higher versus lower permeability in the context of local hydrogeologic formations), and (iii) green text highlights portions of a quote that provide information about the lithology of the formation.

Supplementary Table 29. Hydrostratigraphy details for the Albuquerque

Formation name	Category	Quote
Qu — Alluvium, undivided	Unconsolidated aquifer	" River alluvium; channel, floodplain, and lower terraces deposits of inner Rio Grande and Puerco valleys; as much as 120 ft thick. Map unit "Qf" (Kelley 1977). Forms upper part of the " shallow aquifer " system. Hydrogeologic (lithofacies) subdivision Iv*. Age: Holocene to late Pleistocene. " (Hawley et al., 1994).
Qr — Rio Grande fluvial deposits, undivided	Unconsolidated aquifer	From (Connel, 2006), Plate 1, Explanation of Map Units " Las Padillas Formation (Qrp, historic-upper Holocene) — fluvial deposits of the Rio Grande: pinkish-gray to grayish-brown sand and pebbly sand with lenses of reddish-brown silt and clay; contains paleochannel, point-bar, and overbank levee deposits ; underlies modern (inner) valley of the Rio Grande; very weak to no soil development; base not exposed; 15-34 m thick in wells ; proposed by Connell and Love (2001); divided into 8 units based largely on surface morphology (Kelson et al., 1999): ". "The other major hydrostratigraphic unit (RA) comprises Rio Grande and Puerco deposited of late Quaternary age (<15,000) that form the upper part of the regional shallow-aquifer system. " (Hawley and Haase, 1992).
Ts — Santa Fe Group, undivided middle and lower subgroup	Sedimentary rock aquitard (consolidated or semi-consolidated rock)	"The lower and middle parts of the Santa Fe Group are locally well indurated and contain a large amount of fine- to medium-grained elastic material (clay, silt, and fine sand). " (Hawley et al., 1994). "With the exception of locally extensive and thick eolian sand deposits (e.g., Zia Fm in the lower Santa Fe Group), lower and middle Santa Fe units do not produce significant amounts of good quality groundwater even though they constitute the bulk of the basin-fill sequence." (Hawley et al., 1995).
QTsp — Sierra Ladrones Formation, piedmont member	Unconsolidated aquifer	From (Connel, 2006), Plate 1, Explanation of Map Units " Sierra Ladrones Formation, piedmont member (QTsp, Miocene(?), Pliocene-lower Pleistocene) — reddish- to yellowish-brown conglomerate, sandstone, and minor mudstone; contains weakly developed paleosols and upward-fining sequences of gravel, sand, and mud; contains limestone, sandstone and granite clasts; sparse hypabyssal intrusive clasts present at northeastern end of map area; generally poorly consolidated; interfingers with axial-fluvial member (QTsa); 0-600(?) m thick. ". "Unit includes Ceja Member of Kelley (1977), and Sierra Ladrones Formation of Machette (1978a,b) and Lozinsky and Tedford (1991). Forms lower part of "shallow aquifer" below river-floodplain areas, and upper part of basin-fill aquifer in western part of NE and SE Albuquerque well fields." (Hawley and Haase, 1992).
QTsa — Sierra	Unconsolidated aquifer	From (Connel, 2006), Plate 1, Explanation of Map Units " Sierra Ladrones Formation, axial — fluvial member (QTsa, Miocene(?), Pliocene —

Formation name	Category	Quote
Ladrones Formation, axial-fluvial member		lower Pleistocene — light — gray to yellowish — brown sand, pebbly to cobbly sand, and sparse interbedded mud; clasts dominated by rounded orthoquartzite and volcanic rocks; deposits associated with the ancestral Rio Grande; interfingers with piedmont member (QTsp); contains early Irvingtonian mammalian fossils (SB-1, Table 1); base not exposed; estimated thickness more than 300-800 m. "Sand, pebbly sand and silty sand beds in facies III form a major part of the basin-fill aquifer in the central Albuquerque Basin." (Hawley and Haase, 1992). "Unit includes Ceja Member of Kelley (1977), and Sierra Ladrones Formation of Machette (1978a,b) and Lozinsky and Tedford (1991). Forms lower part of "shallow aquifer" below river-floodplain areas, and upper part of basin-fill aquifer in western part of NE and SE Albuquerque well fields." (Hawley and Haase, 1992).
Tcrp — Ceja Formation, Rio Puerco Member	Unconsolidated aquifer	From (Connel, 2006), Plate 1, Explanation of Map Units " Ceja Formation (Tc, Tcrg, Tcau, Tca, Tcs, Pliocene-lowest Pleistocene(?)) — sand, gravel, and mud derived from western and northwestern Albuquerque basin ; disconformably overlies Arroyo Ojito Formation (Ton and Top) and generally coarsens upsection; defined by Kelley (1977); contains Blancan mammalian fossils (SB-2, Table 1); 20-700(?) m thick ; divided into three members and one subunit:". "The other major hydrostratigraphic unit (RA) comprises Rio Grande and Puerco deposited of late Quaternary age (<15,000) that form the upper part of the regional shallow-aquifer system. " (Hawley and Haase, 1992).
Tca — Ceja Formation, Atrisco Members	Unconsolidated aquifer	From (Connel, 2006), Plate 1, Explanation of Map Units " Atrisco Member (Tca, Pliocene) — pink to yellowish-brown sandstone, pebbly sandstone, and mudstone; interpreted to interfinger with Sierra Ladrones Formation (QTsa) to east; rests on Arroyo Ojito Formation (Ton, Top) and Rincones paleosurface; locally subdivided into upper sandy subunit (Tcau); defined by Connell et al. (1998a); 20-600(?) m thick. ". "Unit includes Ceja Member of Kelley (1977), and Sierra Ladrones Formation of Machette (1978a,b) and Lozinsky and Tedford (1991). Forms lower part of "shallow aquifer" below river-floodplain areas, and upper part of basin-fill aquifer in western part of NE and SE Albuquerque well fields." (Hawley and Haase, 1992).
To — Arroyo Ojito Formation, undivided	Sedimentary rock aquifer (consolidated or semi-consolidated rock)	From (Connel, 2006), Plate 1, Explanation of Map Units " Arroyo Ojito Formation (1999) (To, Ton, Tob, Top, upper Miocene) — gravel-bearing fluvial deposits derived from north and northwest of the Albuquerque basin ; defined by Connell et al. (1999) and modified by Connell (in preparation); 437-456 m thick at type section in Arroyo Ojito; divided into three conformable members: ". "Buried arroyo-channel deposits of a large alluvial fan that spread out from the mouth of Tijeras Canyon (facies Vd) form another major hydrogeologic unit (middle and upper Santa Fe; MSF-1 and USF-1) that has greater than average aquifer potential. " (Hawley and Haase, 1992).
Tcc — Cerro Conejo Formation	Sedimentary rock aquifer (consolidated or semi-consolidated rock)	From (Connel, 2006), Plate 1, Explanation of Map Units " Cerro Conejo Formation (Tcc, middle-upper Miocene) — pink to very pale-brown tabular and cross-stratified sandstone with thin to medium bedded mudstone; contains sandstone concretions and volcanic fallout ashes (11.3-10.8, SA-25 through SA28, Table 1), and late Barstovian mammalian fossils (SB-3, SB-4, Table 1); base is probably disconformable with the Zia Formation; top may be disconformable with Arroyo Ojito Formation to west, but interfingers with Navajo Draw Member to east; originally defined as Cerro Conejo Member

Formation name	Category	Quote
		(Zia Formation) by Connell et al. (1999), but elevated to formation rank based on mappability; 245-316 m thick ." "Eolian (Zia) and facies could be at least a local (future) source of groundwater in the far northwestern part of the basin (west and northwest of Rio Rancho)." (Hawley and Haase, 1992).
Tz — Zia Formation, undivided	Sedimentary rock aquifer (consolidated or semi-consolidated rock)	From (Connel, 2006), Plate 1, Explanation of Map Units " Zia Formation (Tz, lower-middle Miocene) — cross-stratified sandstone and mudstone ; unconformably overlies Galisteo and Menefee Formations (Tg, Kvm) and unit of Isleta well #2 (Tis , in subsurface, see Plate 2); defined by Galusha (1966) and Gawne (1981) and contains late Arikareean through Hemmingfordian mammalian fossils (SB-5 , Table 1; Galusha, 1966) and divided into three members: Cañada Pilares Member (Tzr, middle Miocene) — red mudstone and sandstone; discontinuously exposed; defined by Gawne (1981); 8-75 m thick, Chamisa Mesa Member (Tzm, middle Miocene) — pale-brown to light reddish-brown cross-stratified fluvial and eolian sandstone and mudstone; 30-200 m thick. Piedra Parada Member (Tzp, lower-middle Miocene) — whitish-gray to pinkish-gray cross-stratified sandstone, eolian in origin; 70-122 m thick ." "Eolian (Zia) and facies could be at least a local (future) source of groundwater in the far northwestern part of the basin (west and northwest of Rio Rancho)." (Hawley and Haase, 1992).
Tvi — Buried Igneous rocks	Endogenous bedrock	From (Connel, 2008), under the Description of Map and Cross Section Units, " Buried Igneous rocks (Pliocene – Oligocene) Igneous rocks, locally encountered in boreholes ."
Ys — Sandia granite	Endogenous bedrock	From (Connel, 2006), Plate 1, Explanation of Map Units " Sandia granite (Ys, Mesoproterozoic) — pink megacrystic biotite monzogranite and granodiorite; includes zones of sheared megacrystic biotite monzogranite and granodiorite of the Seven Springs shear zone (Yss) just north of trace of Tijeras fault zone ; U-Pb dates on zircon indicate age of crystallization between 1455±12 Ma and 1446±26 Ma ." "The predominant granitic material is most likely derived from the Sandia Granite ." (Hawley and Haase, 1992).

Hawley, J.W., Haase, C.S. (1992) Hydrogeologic framework of the northern Albuquerque Basin. Open-File Report 387:74 New Mexico Bureau of Mines and Mineral Resources, Socorro, NM. Accessed July 9, 2022 via https://geoinfo.nmt.edu/publications/openfile/downloads/300-399/387/ofr_387.pdf

Hawley, J.W., Haase, C. S., Lozinsky, R. P. (1995). An underground view of the Albuquerque Basin. In *The Water Future of Albuquerque and Middle Rio Grande Basin: Proceedings of the 39th Annual New Mexico Water Conference: November 3-4, 1994, Albuquerque, New Mexico* (No. 290, p. 37). New Mexico Water Resources Research Institute, New Mexico State University. Accessed July 10, 2022 via <https://nmwri.nmsu.edu/wp-content/uploads/2015/watcon/proc39/Hawley.pdf>

Connell, S.D. (2006). Preliminary geologic map of the Albuquerque–Rio Rancho metropolitan area and vicinity. *Bernalillo and Sandoval Counties, New Mexico: New Mexico Bureau of Geology and Mineral Resources. Open-file Report, 496(2)*. Plate 1 and Plate 2 accessed July 9, 2022 via <https://geoinfo.nmt.edu/publications/openfile/details.cfm?Volume=496>

Connell, S.D. (2008). *Geologic Map of the Albuquerque-Rio Rancho Metropolitan Area and Vicinity: Bernalillo and Sandoval Counties, New Mexico*. New Mexico Bureau of Geology and Mineral Resources. Accessed July 9, 2022 via https://geoinfo.nmt.edu/publications/openfile/downloads/400-499/496/OFR-496_Plate2_xsects.pdf

3.27 San Luis Valley, Middle Rio Grande

Supplementary Fig. 151. Hydrogeologic cross section. 20 equally spaced transparent pink bars overlie the cross section; each shaded bar depicts the vertical offset from the land surface to the top of the uppermost confining unit or endogenous bedrock.

Supplementary Fig. 152. Vertical variations in the prevalence of wells that have been defined as tapping an unconfined or a confined aquifer by the USGS. The smaller squares represent 10 m depth intervals from the land surface to 100 m; the larger squares represent 20 m intervals from 100 m to 300 m below the land surface.

The San Luis Valley is located in southern Colorado.

(i) A hydrogeologic cross section presented in Fig. 5 by Leonard and Watts (1989) does not depict a clear confining unit within the aquifer system. However, the hydrogeologic cross section in Plate 6 by Repplier et al. (1981) does depict a shallow confining

unit (*note: this cross section is incorrectly labelled in the cited publication as “Raton Basin”; we confirmed that this cross section represents the San Luis Valley (not the Raton Basin) via personal communication with a Senior Hydrogeologist at the Colorado Geological Survey on April 18, 2022). We added a confining unit to the cross section (top left of this page) on the basis of

(ii) We analysed wells within the study area that the USGS has defined as either unconfined or confined. Most (>80%) wells at depths of 140-160 m and at depths exceeding 140 m are defined as tapping a confined aquifer.

Depth to confined conditions: 140-160 m (see (ii) above)

Reference: Leonard, G. J., Watts, K. R. (1989). Hydrogeology and simulated effects of ground-water development on an unconfined aquifer in the Closed Basin Division, San Luis Valley, Colorado. US Geological Survey Water-Resources Investigations Report 87-4284, 47 pp. Accessed April 10, 2022 via <https://pubs.usgs.gov/wri/1987/4284/report.pdf>

Repplier, F.N., Healy, F.C., Longmire, P.A. (1981). Atlas of Ground Water Quality in Colorado. Map Series 16, <https://coloradogeologicalsurvey.org/publications/atlas-ground-water-quality-colorado/>

The table below presents a series of published quotes (see quotation marks denoting text quoted from another publication, which is cited following the quotation marks with the full reference written in full below the table). The leftmost column lists a title of a hydrogeologic formation depicted in the cross section on the previous page. The rightmost column presents a quote from a hydrogeological study (see base of table for citation). The quote has been annotated with colored text to highlight how we categorized each layer (i.e., see categories in the center column in the table). Specifically: (i) blue text highlights portions of a quote that provide insights into the degree of consolidation of the formation, (ii) red text highlights portions of a quote that categorize the formation as an aquifer or an aquitard (i.e., higher versus lower permeability in the context of local hydrogeologic formations), and (iii) green text highlights portions of a quote that provide information about the lithology of the formation.

Supplementary Table 30. Hydrostratigraphy details for the San Luis Valley

Formation name	Category	Quote
Alamosa Formation (Clay layer exists)	Unconsolidated aquifer	"The Alamosa Formation and overlying deposits consist of discontinuous beds of clay, silt, sand, and gravel. " (Leonard and Watts, 1987). "Alamosa Formation has a maximum thickness of about 2,050 ft in the topographic low (Burroughs, 1981)." Leonard and Watts (1987). "The valley-fill deposits of the San Luis Valley form aquifers that contain ground water. " (Leonard and Watts, 1987).
Santa Fe Formation	Unconsolidated aquifer	"The Santa Fe Formation of Miocene and Pliocene age consists of buff to pinkish-orange clays with interbedded, poorly to moderately sorted silty sands. " (Leonard and Watts (1987). "all 'of the Santa Fe, Los Pinos, Oligocene (?) volcanics, Vallejo and older volcanics and volcanics are lumped into a single aquifer, the "confined." " (Burroughs, 1981).
Los Pinos Formation	Unconsolidated aquifer	"the Los Pinos Formation consists of sandy gravel with interbedded volcaniclastic sandstone and tuffaceous material that was deposited as an eastward thickening wedge along the eastern border of the San Juan Mountains." Leonard and Watts (1987) "all 'of the Santa Fe, Los Pinos, Oligocene (?) volcanics, Vallejo and older volcanics and volcanics are lumped into a single aquifer, the "confined." " (Burroughs, 1981).
Carpenter Ridge and Fish Canyon Tuffs	Volcanic aquifer	"The upper boundary of these volcanics has been placed at the base of the Fish Canyon-Carpenter Ridge ash flow tuffs. These tuffs are at the base of and are interbedded, with the Los Pinos sands (Lipman, 1975)." (Burroughs, 1981). " The Fish Canyon-Carpenter Ridge ash flow tuffs and the Oligocene (?) volcanics should be the main reservoir rocks for geothermal water in the Monte Vista graben." (Burroughs, 1981).
Volcaniclastic and volcanic rocks	Volcanic aquifer	"The basal interval consists of older volcanic and volcaniclastic rocks " " The Fish Canyon-Carpenter Ridge ash flow tuffs and the Oligocene (?) volcanics should be the main reservoir rocks for geothermal water in the Monte Vista graben." (Burroughs, 1981).

Leonard, G.J., Watts, K.R. (1989). Hydrogeology and simulated effects of ground-water development on an unconfined aquifer in the Closed Basin Division, San Luis Valley, Colorado. US Geological Survey No. 87-4284. Accessed June 9, 2022 via <https://pubs.usgs.gov/wri/1987/4284/report.pdf>

Burroughs, R.L. (1981). Summary of the geology of the San Luis Basin, Colorado-New Mexico with emphasis on the geothermal potential for the Monte Vista Graben. Special Publication 17. United States doi:10.2172/6763319.

3.28 Central Mississippi Embayment, Mississippi Embayment

Supplementary Fig. 153. Hydrogeologic cross section. 20 equally spaced transparent pink bars overlie the cross section; each shaded bar depicts the vertical offset from the land surface to the top of the uppermost confining unit or endogenous bedrock.

Supplementary Fig. 154. Vertical variations in the prevalence of wells that have been defined as tapping an unconfined or a confined aquifer by the USGS. The smaller squares represent 10 m depth intervals from the land surface to 100 m; the larger squares represent 20 m intervals from 100 m to 300 m below the land surface.

The Central Mississippi Embayment is located near the Mississippi River in Arkansas, Louisiana, Mississippi and Tennessee.

(i) A hydrogeologic cross section presented in Figs. 68 and 69 by Renken (1998) suggests that the uppermost confining unit (named the Cook Mountain Formation, in the area where the cross section is located) is typically 75 meters below land surface (median vertical offset of the 20 pink transparent bars in the cross section to the left; the 25th-75th percentile range is 49-94 meters below land surface).

(ii) We analysed wells within the study area that the USGS has defined as either unconfined or confined. Most (>80%) wells at depths of 70-80 m and at depths exceeding 70 m are defined as tapping a confined aquifer. This estimate is consistent with the median depth to the top of the uppermost confining layer shown in the hydrogeologic cross section (see (i) above).

(iii) Renken (1998) state that (following text quoted directly): "The alluvial aquifers consist of gravel and sand deposits of Quaternary age and generally contain ground water under unconfined conditions".

Depth to confined conditions: 70-80 m (see (i) and (ii) above)

References: Renken, R. A. (1998). Groundwater Atlas of the United States: Arkansas, Louisiana, Mississippi (HA-730-F). US Geological Survey. https://pubs.usgs.gov/ha/ha730/ch_f/

The table below presents a series of published quotes (see quotation marks denoting text quoted from another publication, which is cited following the quotation marks with the full reference written in full below the table). The leftmost column lists a title of a hydrogeologic formation depicted in the cross section on the previous page. The rightmost column presents a quote from a hydrogeological study (see base of table for citation). The quote has been annotated with colored text to highlight how we categorized each layer (i.e., see categories in the center column in the table). Specifically: (i) blue text highlights portions of a quote that provide insights into the degree of consolidation of the formation, (ii) red text highlights portions of a quote that categorize the formation as an aquifer or an aquitard (i.e., higher versus lower permeability in the context of local hydrogeologic formations), and (iii) green text highlights portions of a quote that provide information about the lithology of the formation.

Supplementary Table 31. Hydrostratigraphy details for the Mississippi Embayment Aquifer System

Formation name	Category	Quote
Mississippi River Valley alluvial aquifer	Unconsolidated aquifer	The alluvial deposits that make up the aquifer are mostly of Holocene age." (Hosman, & Weiss, (1991) "The materials constituting the aquifer range in size from coarse gravel to clay . They commonly grade downward from fine sand, silt, and clay at the top to coarse sand or gravel at the base. " (Hosman, & Weiss, (1991)
Upper Claiborne aquifer	Clastic sedimentary rock aquifer (consolidated or semi-consolidated rock)	"The aquifer consists of interbedded fine sand, silt, and clay with common occurrences of lignite. " (Hosman, & Weiss, (1991)
Cook Mountain Formation	Sedimentary rock aquitard (consolidated or semi-consolidated rock)	"The clay , which is mostly the Cook Mountain Formation (included in the Laredo Formation in part of southern Texas), underlies about 92,000 mi ² of the area." (Hosman, & Weiss, (1991) "the Cook Mountain generally constitutes a confining bed " (Hosman, & Weiss, (1991)
Middle Claiborne aquifer (Sparta Sand)	Clastic sedimentary rock aquifer (consolidated or semi-consolidated rock)	"This aquifer underlies about 136,000 mi ² and consists primarily of the Sparta Sand , which is present in most of the study area as a continentally derived sand with clay interbeds of varying thickness and extent. " (Hosman, & Weiss, (1991)
Cane River Formation	Sedimentary rock aquitard (consolidated or semi-consolidated rock)	"In Texas, three formations are equivalent to the Cane River Formation. They are, in ascending order, the Reklaw Formation, the Queen City Sand, and the Weches Formation (except in extreme southern Texas, where the Bigford Formation and El Pico Clay represent these units). The Reklaw Formation, which is mostly clay , is virtually the entire lower Claiborne confining unit east of the Sabine uplift." (Hosman, & Weiss, (1991)
Lower Claiborne-Upper Wilcox aquifer	Clastic sedimentary rock aquifer (consolidated or semi-consolidated rock)	"Most of the aquifer is the Carrizo Sand and its equivalent, the Meridian Sand Member of the Tallahatta Formation (table 1). The Carrizo or Meridian is an extensive sand, commonly massive and unbroken by clay beds, that represents the basal unit of the Claiborne Group." (Hosman, & Weiss, (1991)
Middle Wilcox aquifer	Clastic sedimentary rock aquifer (consolidated or semi-consolidated rock)	"Because the middle Wilcox aquifer is composed chiefly of thin interbedded sand, silt, and clay , it has water-bearing characteristics different from those of typical massive and aquifers. "

Formation name	Category	Quote
Lower Wilcox aquifer	Clastic sedimentary rock aquifer (consolidated or semi-consolidated rock)	"This aquifer only occurs in the Mississippi embayment aquifer system. In the northern part of the Mississippi embayment, a massive sand aquifer, the Fort Pillow Sand of Tennessee, Arkansas, and Missouri (Moore and Brown, 1969), occurs in the lower to middle part of the Wilcox deposits." Hosman & Weiss (1991)
Midway Group	Sedimentary rock aquitard (consolidated or semi-consolidated rock)	"The Midway (Paleocene) confining unit (pi. 18) is a thick confining layer that is the base of the flow system for Tertiary aquifers in most of the study area." (Hosman, & Weiss, (1991) "The Midway consists mostly of dense marine clays, with lesser amounts of calcareous materials in the lower part." Hosman & Weiss (1991)
Nacatoch Sand	Clastic sedimentary rock aquifer (consolidated or semi-consolidated rock)	"An important aquifer in the northeastern part of the study area, it is composed of sand beds in the Nacatoch Sand in Arkansas; the McNairy Sand in Missouri, Illinois, Kentucky, and Tennessee; and the McNairy Sand Member of the Ripley Formation of northern Mississippi." Hosman & Weiss (1991)
Ozark Plateaus aquifer system	Carbonate aquifer	"Flat-lying to southward-dipping limestone, dolomite, and sandstone comprise the principal aquifers of the Ozark Plateaus aquifer system" Renken (1998)

Hosman, R. L., Weiss, J. S. (1991). Geohydrologic units of the Mississippi embayment and Texas coastal uplands aquifer systems, south-central United States (No. 1416-B). US Government Printing Office. <https://pubs.er.usgs.gov/publication/pp1416B>

Renken, R. A. (1998). Ground Water Atlas of the United States: Segment 5, Arkansas, Louisiana, Mississippi (No. 730-F, pp. F1-F28). US Geological Survey. <https://pubs.er.usgs.gov/publication/ha730F>

3.29 Eastern Mississippi Embayment, Mississippi Embayment

Supplementary Fig. 155. Hydrogeologic cross section. 20 equally spaced transparent pink bars overlie the cross section; each shaded bar depicts the vertical offset from the land surface to the top of the uppermost confining unit or endogenous bedrock.

Supplementary Fig. 156. Vertical variations in the prevalence of wells that have been defined as tapping an unconfined or a confined aquifer by the USGS. The smaller squares represent 10 m depth intervals from the land surface to 100 m; the larger squares represent 20 m intervals from 100 m to 300 m below the land surface.

The Eastern Mississippi Embayment is located in central Mississippi and western Tennessee.

(i) A hydrogeologic cross section presented in Figs. 68 and 69 by Renken (1998) depicts dipping sedimentary sequences including layers classified as aquifers and layers classified as aquitards.

(ii) We analysed wells within the study area that the USGS has defined as either unconfined or confined. Most (>80%) wells at depths of 70-80 m and at depths exceeding 70 m are defined as tapping a confined aquifer.

Depth to confined conditions: 70-80 m (see (ii) above)

Reference: Renken, R. A. (1998). Groundwater Atlas of the United States: Arkansas, Louisiana, Mississippi (HA-730-F). US Geological Survey. https://pubs.usgs.gov/ha/ha730/ch_f/

The table below presents a series of published quotes (see quotation marks denoting text quoted from another publication, which is cited following the quotation marks with the full reference written in full below the table). The leftmost column lists a title of a hydrogeologic formation depicted in the cross section on the previous page. The rightmost column presents a quote from a hydrogeological study (see base of table for citation). The quote has been annotated with colored text to highlight how we categorized each layer (i.e., see categories in the center column in the table). Specifically: (i) blue text highlights portions of a quote that provide insights into the degree of consolidation of the formation, (ii) red text highlights portions of a quote that categorize the formation as an aquifer or an aquitard (i.e., higher versus lower permeability in the context of local hydrogeologic formations), and (iii) green text highlights portions of a quote that provide information about the lithology of the formation.

Supplementary Table 32. Hydrostratigraphy details for the Eastern Mississippi Embayment

Formation name	Category	Quote
Mississippi River Valley alluvial aquifer	Unconsolidated aquifer	The alluvial deposits that make up the aquifer are mostly of Holocene age." (Hosman, & Weiss, (1991) "The materials constituting the aquifer range in size from coarse gravel to clay . They commonly grade downward from fine sand, silt, and clay at the top to coarse sand or gravel at the base. " (Hosman, & Weiss, (1991)
Upper Claiborne aquifer	Clastic sedimentary rock aquifer (consolidated or semi-consolidated rock)	"The aquifer consists of interbedded fine sand, silt, and clay with common occurrences of lignite. " (Hosman, & Weiss, (1991)
Cook Mountain Formation	Sedimentary rock aquitard (consolidated or semi-consolidated rock)	"The clay , which is mostly the Cook Mountain Formation (included in the Laredo Formation in part of southern Texas), underlies about 92,000 mi ² of the area." (Hosman, & Weiss, (1991) "the Cook Mountain generally constitutes a confining bed " (Hosman, & Weiss, (1991)
Middle Claiborne aquifer (Sparta Sand)	Clastic sedimentary rock aquifer (consolidated or semi-consolidated rock)	"This aquifer underlies about 136,000 mi ² and consists primarily of the Sparta Sand , which is present in most of the study area as a continentally derived sand with clay interbeds of varying thickness and extent. " (Hosman, & Weiss, (1991)
Cane River Formation	Sedimentary rock aquitard (consolidated or semi-consolidated rock)	"In Texas, three formations are equivalent to the Cane River Formation. They are, in ascending order, the Reklaw Formation, the Queen City Sand, and the Weches Formation (except in extreme southern Texas, where the Bigford Formation and El Pico Clay represent these units). The Reklaw Formation, which is mostly clay , is virtually the entire lower Claiborne confining unit east of the Sabine uplift." (Hosman, & Weiss, (1991)
Lower Claiborne-Upper Wilcox aquifer	Clastic sedimentary rock aquifer (consolidated or semi-consolidated rock)	"Most of the aquifer is the Carrizo Sand and its equivalent, the Meridian Sand Member of the Tallahatta Formation (table 1). The Carrizo or Meridian is an extensive sand, commonly massive and unbroken by clay beds, that represents the basal unit of the Claiborne Group." (Hosman, & Weiss, (1991)
Middle Wilcox aquifer	Clastic sedimentary rock aquifer (consolidated or semi-consolidated rock)	"Because the middle Wilcox aquifer is composed chiefly of thin interbedded sand, silt, and clay , it has water-bearing characteristics different from those of typical massive and aquifers. "

Formation name	Category	Quote
Lower Wilcox aquifer	Clastic sedimentary rock aquifer (consolidated or semi-consolidated rock)	"This aquifer only occurs in the Mississippi embayment aquifer system. In the northern part of the Mississippi embayment, a massive sand aquifer, the Fort Pillow Sand of Tennessee, Arkansas, and Missouri (Moore and Brown, 1969), occurs in the lower to middle part of the Wilcox deposits." (Hosman, & Weiss, (1991)
Midway Group	Sedimentary rock aquitard (consolidated or semi-consolidated rock)	"The Midway (Paleocene) confining unit (pi. 18) is a thick confining layer that is the base of the flow system for Tertiary aquifers in most of the study area." (Hosman, & Weiss, (1991) "The Midway consists mostly of dense marine clays, with lesser amounts of calcareous materials in the lower part." (Hosman, & Weiss, (1991)
Nacatoch Sand	Clastic sedimentary rock aquifer (consolidated or semi-consolidated rock)	"An important aquifer in the northeastern part of the study area, it is composed of sand beds in the Nacatoch Sand in Arkansas; the McNairy Sand in Missouri, Illinois, Kentucky, and Tennessee; and the McNairy Sand Member of the Ripley Formation of northern Mississippi." (Hosman, & Weiss, (1991)
Ozark Plateaus aquifer system	Carbonate aquifer	"Flat-lying to southward-dipping limestone, dolomite, and sandstone comprise the principal aquifers of the Ozark Plateaus aquifer system" Renken, R. A. (1998)

Renken, R. A. (1998). Ground Water Atlas of the United States: Segment 5, Arkansas, Louisiana, Mississippi (No. 730-F, pp. F1-F28). US Geological Survey. <https://pubs.er.usgs.gov/publication/ha730F>

3.30 Delmarva Peninsula, North Atlantic Coastal Plain

Supplementary Fig. 157. Hydrogeologic cross section. 20 equally spaced transparent pink bars overlie the cross section; each shaded bar depicts the vertical offset from the land surface to the top of the uppermost confining unit or endogenous bedrock.

Supplementary Fig. 158. Vertical variations in the prevalence of wells that have been defined as tapping an unconfined or a confined aquifer by the USGS. The smaller squares represent 10 m depth intervals from the land surface to 100 m; the larger squares represent 20 m intervals from 100 m to 300 m below the land surface.

The Delmarva Peninsula is located in eastern Maryland and Delaware.

(i) A hydrogeologic cross section presented in Fig. 21 of Sanford et al. (2012) depicts a thin (less than ~50 m) shallow aquifer underlain by a layered aquifer system consisting of aquitards and aquifers.

(ii) We analysed wells within the study area that the USGS has defined as either unconfined or confined. Most (>80%) wells at depths of 30-40 m and at depths exceeding 30 m are defined as tapping a confined aquifer.

Depth to confined conditions: 30-40 meters below land surface (based on (ii) above)

Reference: Sanford, W. E., Pope, J. P., Selnick, D. L., Stumvoll, R. F. (2012). Simulation of groundwater flow in the shallow aquifer system of the Delmarva Peninsula, Maryland and Delaware. US Geological Survey Open-File Report 2012-1140. 68 pp. Accessed November 29, 2021 from https://pubs.usgs.gov/of/2012/1140/pdf/OFR_2012-1140.pdf

The table below presents a series of published quotes (see quotation marks denoting text quoted from another publication, which is cited following the quotation marks with the full reference written in full below the table). The leftmost column lists a title of a hydrogeologic formation depicted in the cross section on the previous page. The rightmost column presents a quote from a hydrogeological study (see base of table for citation). The quote has been annotated with colored text to highlight how we categorized each layer (i.e., see categories in the center column in the table). Specifically: (i) blue text highlights portions of a quote that provide insights into the degree of consolidation of the formation, (ii) red text highlights portions of a quote that categorize the formation as an aquifer or an aquitard (i.e., higher versus lower permeability in the context of local hydrogeologic formations), and (iii) green text highlights portions of a quote that provide information about the lithology of the formation.

Supplementary Table 33. Hydrostratigraphy details for the Delmarva Peninsula

Formation name	Category	Quote
Surficial aquifer	Unconsolidated aquifer	"The surficial aquifer is composed of a veneer of Upper Miocene to Holocene age sediments that mantle Cretaceous and older Tertiary sediment in Maryland and Delaware." (Vroblesky and Fleck, 1991). "Otton (1955, p.104) has divided the lowland deposits into three lithologic units a basal sand and gravel, an intermediate tough clay, and an upper bed or beds of sandy clay or clayey gravel. Diatoms, marine shells, plant debris, and vivianite are common in the clay. " (Vroblesky and Fleck, 1991).
Upper Chesapeake confining unit UC1	Sedimentary rock aquitard (consolidated or semi-consolidated rock)	"The upper Chesapeake confining unit – The upper Chesapeake confining unit (fig. 27), in the uppermost part of the Chesapeake Group, is a discontinuous unit of lenticular silt, clay, and fine sand separating the Upper Chesapeake aquifer from the overlying surficial aquifer downdip from the subcrop area." (Vroblesky and Fleck, 1991).
Pocomoke aquifer	Sedimentary rock aquifer (consolidated or semi-consolidated rock)	"The upper Chesapeake aquifer contains three major sand bodies . They are, from lowermost to uppermost, the Pocomoke aquifer, the Ocean City aquifer, and the Manokin aquifer (Weigle, 1974, p. 31-33; Hansen, 1981b)." (Vroblesky and Fleck, 1991). "The Pocomoke aquifer consists of gray, fine- to medium-grained sand and some interbedded silt and clay. " (Vroblesky and Fleck, 1991).
Upper Chesapeake confining unit UC2	Sedimentary rock aquitard (consolidated or semi-consolidated rock)	"The upper Chesapeake confining unit – The upper Chesapeake confining unit (fig. 27), in the uppermost part of the Chesapeake Group, is a discontinuous unit of lenticular silt, clay, and fine sand separating the Upper Chesapeake aquifer from the overlying surficial aquifer downdip from the subcrop area." (Vroblesky and Fleck, 1991).
Manokin aquifer	Sedimentary rock aquifer (consolidated or semi-consolidated rock)	"The Manokin aquifer is composed of the same general material and may contain coarse sand and pea-sized gravel in basal units . The Pocomoke and Manokin aquifers are separated by a sequence of clay, silt, and fine sand , ranging in thickness from 20 to 50 ft in Delaware (Miller, 1971, p. 14, 16) (Vroblesky and Fleck, 1991).
St. Marys confining unit	Sedimentary rock aquitard (consolidated or semi-consolidated rock)	"The regional confining unit overlying the Lower Chesapeake aquifer includes an unnamed confining unit in New Jersey, the Saint Marys confining unit in Delaware, Maryland, and Virginia, and the Pungo River confining unit in North Carolina. This Miocene unit primarily composes silt and clay but is diatomaceous in New Jersey and silty and

Formation name	Category	Quote
		shelly in Delaware, Maryland, and Virginia (Trapp, 1992)." (Masterson et al., 2015).
Choptank aquifer	Sedimentary rock aquifer (consolidated or semi-consolidated rock)	" The lower Chesapeake aquifer (fig. 24) is that part of the Miocene Calvert and Choptank Formations in the Chesapeake Group that is sandy enough to function as an aquifer. " (Vroblesky and Fleck, 1991). "The sediments of the lower Chesapeake aquifer consist of medium to coarse silty sand and clay having locally abundant shells. " (Vroblesky and Fleck, 1991).
Lower Chesapeake confining unit	Sedimentary rock aquitard (consolidated or semi-consolidated rock)	"The lower Chesapeake confining unit consists of the silt, clay, fine sand, and diatomaceous earth between the Piney Point-Nanjemoy aquifer and the overlying lower Chesapeake aquifer. Point-Nanjemoy and Aquia-Rancocas aquifers beyond the updip limit of the lower Chesapeake aquifer. In eastern Maryland, the confining unit typically consists of clayey beds of the lowermost part of the Miocene Chesapeake Group (Otton, 1955, p. 90-95)."
Calvert aquifer	Clastic sedimentary rock aquifer (consolidated or semi-consolidated rock)	"The Lower Chesapeake regional aquifer includes the Lower Kirkwood-Cohansey aquifer system in New Jersey, the Milford, Frederica, Federalsburg, and Cheswold local aquifers in Delaware, the Choptank and Calvert local aquifers in Maryland, the Saint Marys aquifer in Virginia, and the Pungo River aquifer in North Carolina." (Masterson et al., 2013). " The Lower Chesapeake aquifer consists primarily of marine sands ranging in age from Oligocene to Pliocene. In New Jersey, the aquifer includes interbedded sand and gravel (Trapp, 1992)." (Masterson et al., 2015).
Calvert confining unit	Sedimentary rock aquitard (consolidated or semi-consolidated rock)	"It consists primarily of marine clay and sandy clay of Miocene age, and its thickness increases northward. " (Masterson et al., 2013). "The confining unit overlying the Piney Point regional aquifer and separating it from the Lower Chesapeake aquifer above includes the Basal Kirkwood confining unit in New Jersey, the Calvert confining unit in Delaware, Maryland, and Virginia, and the Castle Hayne confining unit in North Carolina. It consists primarily of marine clay and sandy clay of Miocene age, and its thickness increases northward from less than 50 ft in North Carolina to a range of 100 to 250 ft over the northern half of the study area (Trapp, 1992)." (Masterson et al., 2013).
Piney Point aquifer	Sedimentary rock aquifer (consolidated or semi-consolidated rock)	"The Piney Point aquifer consists of marine sediments of mostly Eocene to Oligocene age, though it also may include sediments of Miocene age in some locations " (Masterson et al., 2013) "a productive section of the Piney Point aquifer north of the James River and south of the Potomac River in Virginia composes calcite-cemented sands and moldic limestone. " (Masterson et al., 2013)
Nanjemoy confining unit	Sedimentary rock aquitard (consolidated or semi-consolidated rock)	"A rapid marine transgression followed and is marked by the basal clay and silt beds of the Eocene Nanjemoy Formation." (Vroblesky and Fleck, 1991). " The Nanjemoy-Marlboro confining unit. The Nanjemoy-Marlboro confining unit is typically the clayey material between the sharp upper contact of the Aquia- Rancocas aquifer and the gradational lower contact of the Piney Point-Nanjemoy aquifer." (Vroblesky and Fleck, 1991).

Formation name	Category	Quote
Aquia aquifer	Sedimentary rock aquifer (consolidated or semi-consolidated rock)	"The Aquia aquifer composes permeable marine sediments of Paleocene age and consists primarily of medium- to coarse grained glauconitic and fossiliferous quartz sands. " (Masterson et al., 2013)
Severn confining unit	Sedimentary rock aquitard (consolidated or semi-consolidated rock)	"The northern section of the Monmouth-Mount Laurel regional aquifer is separated from the Aquia regional aquifer above by an overlying confining unit that includes the Navesink-Hornerstown confining unit in New Jersey and the Severn confining unit in Delaware and Maryland. The confining unit consists of marine silt, clay, and silty and clayey glauconitic sand of primarily Cretaceous age. " (Masterson et al., 2013).
Monmouth aquifer	Sedimentary rock aquifer (consolidated or semi-consolidated rock)	"The Monmouth-Mount Laurel regional aquifer (previously referred to by Trapp and Meisler (1992) as the Peedee-Severn aquifer) includes the Wenonah-Mount Laurel aquifer in New Jersey, the Mount Laurel aquifer in Delaware, and the Monmouth aquifer in Maryland (fig. 9)." (Masterson et al., 2013). "The Monmouth-Mount Laurel aquifer is the uppermost regional aquifer of Late Cretaceous age in the study area. It includes permeable parts of the marine Mount Laurel Formation , which is the lower part of the Monmouth Group (Andreasen and others, 2013). This aquifer consists of very fine to coarse, slightly glauconitic sand in New Jersey and fine glauconitic sand in Delaware and Maryland and ranges in thickness from about 10 to 120 ft in Delaware and Maryland and in southern New Jersey (Zapeczka, 1989; Andreasen and others, 2013)." (Masterson et al., 2013).
Matawan confining unit	Sedimentary rock aquifer (consolidated or semi-consolidated rock)	" The Matawan confining unit – The Magothy aquifer grades upward into the glauconitic clay and silt of the Matawan confining unit. " (Vroblesky and Fleck, 1991). "The Patapsco confining unit – The Patapsco aquifer is overlain by the red, plastic clay of the Patapsco confining unit in the upper part of the Patapsco Formation." (Vroblesky and Fleck, 1991).
Matawan-Magothy confining unit	Sedimentary rock aquitard (consolidated or semi-consolidated rock)	"Except on Long Island, the Magothy regional aquifer is overlain by a confining unit that separates it from the overlying Matawan aquifer. This regional confining unit includes the Merchantville-Woodbury confining unit in New Jersey, consisting of glauconitic and micaceous clay and silt, and the Matawan-Magothy confining unit in Delaware and Maryland, consisting of silt and clay of the Magothy Formation and part of the overlying Matawan Group or Matawan Formation (Trapp, 1992; Andreasen and others, 2013)." (Masterson et al., 2013)
Magothy aquifer	Sedimentary rock aquifer (consolidated or semi-consolidated rock)	"The Magothy aquifer composes primarily sandy parts of the Magothy Formation, which were deposited in a transitional fluvial-marine environment during the Late Cretaceous. " (Masterson et al., 2013)
Magothy-Patapsco confining unit	Sedimentary rock aquitard (consolidated or	"This unit is referred to as the Raritan Clay confining unit in New York and New Jersey, the Magothy-Patapsco confining unit in Delaware and Maryland." (Masterson et al., 2013)

Formation name	Category	Quote
	semi-consolidated rock)	
Upper Patapsco aquifer	Sedimentary rock aquifer (consolidated or semi-consolidated rock)	"The Potomac-Patapsco regional aquifer includes the Lloyd aquifer in New York (Long Island), the middle aquifer of the Potomac-Raritan-Magothy aquifer system in New Jersey, the Upper and Lower Patapsco aquifers of Delaware and Maryland, the undifferentiated Potomac aquifer of Virginia, and the Lower Cretaceous aquifer in North Carolina (fig. 9)." (Masterson et al., 2013). "The Potomac-Patapsco aquifer is similar to the underlying Potomac-Patuxent aquifer in that it consists primarily of lenses of medium- to coarse-grained quartz sand with some gravel, interbedded with lenses of clay and silt. " (Masterson et al., 2013). "It is composed of fluvial-deltaic sediments of primarily Early Cretaceous age in Maryland and Delaware, and Late Cretaceous age in New Jersey and New York." (Masterson et al., 2013). "The top of the Potomac aquifer in Virginia now correlates with the top of the Upper Patapsco aquifer in Maryland" (Masterson et al., 2013). "It is composed of fluvial-deltaic sediments of primarily Early Cretaceous age in Maryland and Delaware, and Late Cretaceous age in New Jersey and New York." (Masterson et al., 2013).
Patapsco confining unit	Sedimentary rock aquitard (consolidated or semi-consolidated rock)	"The Patapsco confining unit – The Patapsco aquifer is overlain by the red, plastic clay of the Patapsco confining unit in the upper part of the Patapsco Formation." (Vroblesky and Fleck, 1991).
Lower Patapsco aquifer	Sedimentary rock aquifer (consolidated or semi-consolidated rock)	"The Potomac-Patapsco regional aquifer includes the Lloyd aquifer in New York (Long Island), the middle aquifer of the Potomac-Raritan-Magothy aquifer system in New Jersey, the Upper and Lower Patapsco aquifers of Delaware and Maryland, the undifferentiated Potomac aquifer of Virginia, and the Lower Cretaceous aquifer in North Carolina (fig. 9)." (Masterson et al., 2013). "The Potomac-Patapsco aquifer is similar to the underlying Potomac-Patuxent aquifer in that it consists primarily of lenses of medium- to coarse-grained quartz sand with some gravel, interbedded with lenses of clay and silt. " (Masterson et al., 2013). "It is composed of fluvial-deltaic sediments of primarily Early Cretaceous age in Maryland and Delaware, and Late Cretaceous age in New Jersey and New York." (Masterson et al., 2013).

Masterson, J. P., Pope, J. P., Monti Jr, J., Nardi, M. R., Finkelstein, J. S., McCoy, K. J. (2013). Hydrogeology and hydrologic conditions of the Northern Atlantic Coastal Plain aquifer system from Long Island, New York, to North Carolina (No. 2013-5133). US Geological Survey. Accessed July 6, 2022 via <https://pubs.er.usgs.gov/publication/sir20135133>

Vroblesky, D.A., Fleck, W.B. (1991). Hydrogeologic Framework of the Coastal Plain of Maryland, Delaware, and the District of Columbia. U.S. Geological Survey Professional Paper 1404-E, 52 pp. Accessed April 1, 2021 from <https://pubs.usgs.gov/pp/1404e/report.pdf>

3.31 Maryland Western Shores, North Atlantic Coastal Plain

Supplementary Fig. 159. Hydrogeologic cross section. 20 equally spaced transparent pink bars overlie the cross section; each shaded bar depicts the vertical offset from the land surface to the top of the uppermost confining unit or endogenous bedrock.

Supplementary Fig. 160. Vertical variations in the prevalence of wells that have been defined as tapping an unconfined or a confined aquifer by the USGS. The smaller squares represent 10 m depth intervals from the land surface to 100 m; the larger squares represent 20 m intervals from 100 m to 300 m below the land surface.

The Maryland Western Shores area is located in central Maryland.

(i) A hydrogeologic cross section presented in Figs. 4-6 by Vroblesky et al. (1991) depicts a shallow geologic formation with both permeable and less-permeable subunits (undivided).

(ii) We analysed wells within the study area that the USGS has defined as either unconfined or confined. Most (>80%) wells at depths of 30-40 m and at depths exceeding 30 m are defined as tapping a confined aquifer.

Depth to confined conditions: 30-40 meters below land surface (based on (ii) above)

Reference: Vroblesky, D.A., Fleck, W.B. (1991). Hydrogeologic Framework of the Coastal Plain of Maryland, Delaware, and the District of Columbia. U.S. Geological Survey Professional Paper 1404-E, 52 pp. Accessed April 1, 2021 from <https://pubs.usgs.gov/pp/1404e/report.pdf>

The table below presents a series of published quotes (see quotation marks denoting text quoted from another publication, which is cited following the quotation marks with the full reference written in full below the table). The leftmost column lists a title of a hydrogeologic formation depicted in the cross section on the previous page. The rightmost column presents a quote from a hydrogeological study (see base of table for citation). The quote has been annotated with colored text to highlight how we categorized each layer (i.e., see categories in the center column in the table). Specifically: (i) blue text highlights portions of a quote that provide insights into the degree of consolidation of the formation, (ii) red text highlights portions of a quote that categorize the formation as an aquifer or an aquitard (i.e., higher versus lower permeability in the context of local hydrogeologic formations), and (iii) green text highlights portions of a quote that provide information about the lithology of the formation.

Supplementary Table 34. Hydrostratigraphy details for the Maryland Western Shores

Formation name	Category	Quote
Aquifers and confining unit undivided	Unconsolidated aquifer	"The surficial aquifer is composed of a veneer of Upper Miocene to Holocene age sediments that mantle Cretaceous and older Tertiary sediment in Maryland and Delaware." Vroblesky (1991). "Otton (1955, p. 104) has divided the lowland deposits into three lithologic units a basal sand and gravel, an intermediate tough clay, and an upper bed or beds of sandy clay or clayey gravel. " Vroblesky (1991). "The St. Marys confining unit (fig. 25), which overlies the lower Chesapeake aquifer, is composed of gray clay, clayey silt, and very fine sand of the Miocene St. Marys Formation in the middle part of the Chesapeake Group." Vroblesky (1991). "The upper Chesapeake aquifer contains three major sand bodies. " Vroblesky (1991). "The lower Chesapeake aquifer is a multilayer aquifer." Vroblesky (1991). "it is difficult to distinguish the lower Chesapeake aquifer from the overlying surficial aquifer in subcrop areas because of similar lithologies, although the lower Chesapeake aquifer sediments have been reported to be grayer and better sorted (Sundstrom and Pickett, 1969, p. 17-20)." Vroblesky (1991).
Piney Point-Nanjemoy aquifer	Sedimentary rock aquifer (consolidated or semi-consolidated rock)	"The Piney Point aquifer consists of marine sediments of mostly Eocene to Oligocene age, though it also may include sediments of Miocene age in some locations" (Masterson et al., 2013) "a productive section of the Piney Point aquifer north of the James River and south of the Potomac River in Virginia composes calcite-cemented sands and moldic limestone. " (Masterson et al., 2013)
Nanjemoy-Marlboro confining unit	Sedimentary rock aquitard (consolidated or semi-consolidated rock)	"The regional confining unit includes the Vincentown Manasquan confining unit in New Jersey; the Nanjemoy-Marlboro Clay confining unit in Delaware, Maryland, and Virginia; and the Beaufort confining unit in North Carolina." (Masterson et al., 2013) "The confining unit overlaying the Aquia aquifer is made of marine silt, clay, and sandy clay ranging from a thickness of 50 ft in North Carolina to more than 900 ft in New Jersey (Trapp, 1992)." (Masterson et al., 2013)
Aquia-Rancocas aquifer	Sedimentary rock aquifer (consolidated or semi-consolidated rock)	"The Aquia-Rancocas aquifer is composed of the sandy portions of the Paleocene Aquia Formation in Maryland and the Paleocene Rancocas Group in Delaware. In places, it may include sandy portions of the underlying Brightseat Formation." Vroblesky (1991). "The Aquia-Rancocas aquifer is predominantly glauconitic and quartzose, medium- to coarse-grained, and medium- to well-sorted sand (Chapelle and Drummond, 1983, p. 7)." Vroblesky (1991).
Upper Brightseat	Sedimentary rock aquitard (consolidated)	"The upper Brightseat confining unit – The upper Brightseat confining unit , in what may be the upper part of the Brightseat Formation (Bennett and Collins, 1952), overlies the Brightseat aquifer."

Formation name	Category	Quote
confining unit	or semi-consolidated rock)	Vroblesky (1991). "The confining unit is composed of greenish-gray to black, glauconitic silt and clay, having interbedded glauconitic, fine-grained sand (Weigle and Webb, 1970, p. 32). Vroblesky (1991).
Brightseat aquifer	Sedimentary rock aquifer (consolidated or semi-consolidated rock)	"The Brightseat aquifer is restricted to the southernmost part of the study area. It includes sediments that are located primarily in St. Marys County, Md. , that earlier studies assigned to the Magothy aquifer (Hansen, 1972, p. 47; Weigle and Webb, 1970, p. 32)." Vroblesky (1991). "In St. Marys County, the aquifer is composed of very fine to fine, light-gray to yellowish or purple quartzose sand and muscovite, lignite, and minor glauconite (Weigle and Webb, 1970, p. 32)." Vroblesky (1991).
Lower Brightseat, Severn, Matawan, and Patapsco confining units undivided	Sedimentary rock aquitarid (consolidated or semi-consolidated rock)	"The lower Brightseat confining unit – The lower Brightseat confining unit typically consists of the silt and clay between the Severn aquifer and the overlying Aquia-Rancocas aquifer." Vroblesky (1991). "The Severn confining unit – The Severn confining unit , in the lower part of the Upper Cretaceous Severn Formation in Maryland, is composed of the clay and silt between the Matawan aquifer and the overlying Severn aquifer." Vroblesky (1991). "The Matawan confining unit – The Magothy aquifer grades upward into the glauconitic clay and silt of the Matawan confining unit ." Vroblesky (1991). "The Patapsco confining unit – The Patapsco aquifer is overlain by the red, plastic clay of the Patapsco confining unit in the upper part of the Patapsco Formation." Vroblesky (1991).
Patapsco aquifer	Sedimentary rock aquifer (consolidated or semi-consolidated rock)	"The sediments of the Patapsco aquifer are typically white to yellow, crossbedded, fine to medium, clayey sand and subordinate amounts of gravel. Associated clay is dense, massive or laminated, and variegated in shades of red, gray, brown, and purple (Glaser, 1969, p. 9). The aquifer is predominantly sand in Anne Arundel County." Vroblesky (1991). "The top of the Potomac aquifer in Virginia now correlates with the top of the Upper Patapsco aquifer in Maryland" (Masterson et al., 2013). "It is composed of fluvial-deltaic sediments of primarily Early Cretaceous age in Maryland and Delaware, and Late Cretaceous age in New Jersey and New York." (Masterson et al., 2013).
Potomac confining unit	Sedimentary rock aquitarid (consolidated or semi-consolidated rock)	"The Potomac confining unit – The Potomac confining unit overlies the Patuxent aquifer. Vroblesky (1991). " Potomac confining unit generally corresponds to a zone of clay and sand lenses separating two predominantly sandy zones in the Potomac Group (or Formation), as described by Sundstrom and others (1967, p. 21). Vroblesky (1991).

Masterson, J. P., Pope, J. P., Monti Jr, J., Nardi, M. R., Finkelstein, J. S., McCoy, K. J. (2013). Hydrogeology and hydrologic conditions of the Northern Atlantic Coastal Plain aquifer system from Long Island, New York, to North Carolina (No. 2013-5133). US Geological Survey.

McFarland, E.R., Bruce, T.S. (2006). The Virginia Coastal Plain Hydrogeologic Framework: U.S. Geological Survey Professional Paper 1731, 118 p., 25 pls Accessed February 21, 2022 from <https://pubs.usgs.gov/pp/2006/1731/PP1731.pdf>

Vroblesky, D.A., Fleck, W.B. (1991). Hydrogeologic Framework of the Coastal Plain of Maryland, Delaware, and the District of Columbia. U.S. Geological Survey Professional Paper 1404-E, 52 pp. Accessed April 1, 2021 from <https://pubs.usgs.gov/pp/1404e/report.pdf>

3.32 New Jersey Coastal Plain, North Atlantic Coastal Plain

Supplementary Fig. 161. Hydrogeologic cross section. 20 equally spaced transparent pink bars overlie the cross section; each shaded bar depicts the vertical offset from the land surface to the top of the uppermost confining unit or endogenous bedrock.

Supplementary Fig. 162. Vertical variations in the prevalence of wells that have been defined as tapping an unconfined or a confined aquifer by the USGS. The smaller squares represent 10 m depth intervals from the land surface to 100 m; the larger squares represent 20 m intervals from 100 m to 300 m below the land surface.

The New Jersey Coastal Plain is located in eastern New Jersey.

(i) A hydrogeologic cross section presented in Fig. 10 of Masterson et al. (2013) shows a layered aquifer system including clastic sedimentary aquifers and aquitards of varying thicknesses.

(ii) We analysed wells within the study area that the USGS has defined as either unconfined or confined. Most (>80%) wells at depths of 80-90 m and at depths exceeding 80 m are defined as tapping a confined aquifer.

Depth to confined conditions: 80-90 meters below land surface (based on (ii) above)

Reference: Masterson, J.P., Pope, J.P., Monti, Jack, Jr., Nardi, M.R., Finkelstein, J.S., McCoy, K.J. (2013). Hydrogeology and hydrologic conditions of the Northern Atlantic Coastal Plain aquifer system from Long Island, New York, to North Carolina. US Geological Survey Scientific Investigations Report 2013-5133, 88 pp. Accessed June 13, 2022 via <http://dx.doi.org/10.3133/sir20135133>

The table below presents a series of published quotes (see quotation marks denoting text quoted from another publication, which is cited following the quotation marks with the full reference written in full below the table). The leftmost column lists a title of a hydrogeologic formation depicted in the cross section on the previous page. The rightmost column presents a quote from a hydrogeological study (see base of table for citation). The quote has been annotated with colored text to highlight how we categorized each layer (i.e., see categories in the center column in the table). Specifically: (i) blue text highlights portions of a quote that provide insights into the degree of consolidation of the formation, (ii) red text highlights portions of a quote that categorize the formation as an aquifer or an aquitard (i.e., higher versus lower permeability in the context of local hydrogeologic formations), and (iii) green text highlights portions of a quote that provide information about the lithology of the formation.

Supplementary Table 35. Hydrostratigraphy details for the New Jersey Coastal Plain

Formation name	Category	Quote
Kirkwood-Cohansey aquifer system	Unconsolidated aquifer	" unconfined Kirkwood-Cohansey aquifer is hydrologically equivalent to the Surficial aquifer as described in other States, and therefore could be grouped with the regional Surficial regional aquifer ." (Masterson et al., 2015). " The Upper Chesapeake aquifer includes the Upper Kirkwood-Cohansey aquifer in New Jersey" (Masterson et al., 2015). "This aquifer consists of permeable sediments of the upper part of the Miocene to Pliocene-age Chesapeake Group. The aquifer composes primarily sands of marine origin in North Carolina and Virginia but transitions northward to New Jersey into coarser sands and gravels of fluvial origin (Trapp, 1992)." (Masterson et al., 2015). "The Lower Chesapeake regional aquifer includes the Lower Kirkwood-Cohansey aquifer system in New Jersey". (Masterson et al., 2015). "The Lower Chesapeake aquifer consists primarily of marine sands ranging in age from Oligocene to Pliocene. In New Jersey, the aquifer includes interbedded sand and gravel (Trapp, 1992)." (Masterson et al., 2015).
Confining unit	Sedimentary rock aquitard (consolidated or semi-consolidated rock)	"The regional confining unit overlying the Lower Chesapeake aquifer includes an unnamed confining unit in New Jersey, the Saint Marys confining unit in Delaware, Maryland, and Virginia, and the Pungo River confining unit in North Carolina. This Miocene unit primarily composes silt and clay but is diatomaceous in New Jersey and silty and shelly in Delaware, Maryland, and Virginia (Trapp, 1992)." (Masterson et al., 2015).
Vincentown aquifer	Sedimentary rock aquifer (consolidated or semi-consolidated rock)	"The Aquia regional aquifer (previously referred to by Trapp and Meisler (1992) as the Beaufort-Aquia aquifer) includes the Vincentown aquifer in New Jersey, the Rancocas aquifer in Delaware, the Aquia aquifer in Maryland and Virginia, and the Beaufort aquifer in North Carolina (fig. 9)". (Masterson et al., 2015). "The Aquia aquifer composes permeable marine sediments of Paleocene age and consists primarily of medium- to coarse-grained glauconitic and fossiliferous quartz sands (Trapp, 1992)." (Masterson et al., 2015).
Confining unit	Sedimentary rock aquitard (consolidated or semi-consolidated rock)	"The regional confining unit overlying the Lower Chesapeake aquifer includes an unnamed confining unit in New Jersey, the Saint Marys confining unit in Delaware, Maryland, and Virginia, and the Pungo River confining unit in North Carolina. This Miocene unit primarily composes silt and clay but is diatomaceous in New Jersey and silty and

Formation name	Category	Quote
		shelly in Delaware, Maryland, and Virginia (Trapp, 1992)." (Masterson et al., 2015).
Wenonah-Mount Laurel aquifer	Sedimentary rock aquifer (consolidated or semi-consolidated rock)	"The Monmouth-Mount Laurel regional aquifer (previously referred to by Trapp and Meisler (1992) as the Peedee-Severn aquifer) includes the Wenonah-Mount Laurel aquifer in New Jersey, the Mount Laurel aquifer in Delaware, and the Monmouth aquifer in Maryland (fig. 9)". (Masterson et al., 2015). " The Monmouth-Mount Laurel aquifer is the uppermost regional aquifer of Late Cretaceous age in the study area. It includes permeable parts of the marine Mount Laurel Formation , which is the lower part of the Monmouth Group (Andreasen and others, 2013). This aquifer consists of very fine to coarse, slightly glauconitic sand in New Jersey". (Masterson et al., 2015).
Confining unit	Sedimentary rock aquitard (consolidated or semi-consolidated rock)	"The regional confining unit overlying the Lower Chesapeake aquifer includes an unnamed confining unit in New Jersey, the Saint Marys confining unit in Delaware, Maryland, and Virginia, and the Pungo River confining unit in North Carolina. This Miocene unit primarily composes silt and clay but is diatomaceous in New Jersey and silty and shelly in Delaware, Maryland, and Virginia (Trapp, 1992)." (Masterson et al., 2015).
Englishtown aquifer system	Sedimentary rock aquifer (consolidated or semi-consolidated rock)	" The northern section of the Matawan aquifer includes the Englishtown aquifer in New Jersey " (Masterson et al., 2015). " The Matawan aquifer primarily consists of sands deposited by a marine transgression during the Late Cretaceous , including fine to medium quartz sand in New Jersey and fine silty to clayey sand in Delaware and Maryland.". (Masterson et al., 2015).
Confining unit	Sedimentary rock aquitard (consolidated or semi-consolidated rock)	"The regional confining unit overlying the Lower Chesapeake aquifer includes an unnamed confining unit in New Jersey, the Saint Marys confining unit in Delaware, Maryland, and Virginia, and the Pungo River confining unit in North Carolina. This Miocene unit primarily composes silt and clay but is diatomaceous in New Jersey and silty and shelly in Delaware, Maryland, and Virginia (Trapp, 1992)." (Masterson et al., 2015).
Upper Potomac-Raritan-Magothy aquifer (The aquifer is dominantly confined aquifer)	Sedimentary rock aquifer (consolidated or semi-consolidated rock)	"The local aquifer previously considered part of the Upper Potomac aquifer now are part of the regional Magothy aquifer . These units include the Upper Potomac-Raritan-Magothy aquifer in New Jersey". (Masterson et al., 2015). " The Magothy aquifer composes primarily sandy parts of the Magothy Formation , which were deposited in a transitional fluvial-marine environment during the Late Cretaceous (Andreasen and others, 2013). The Magothy aquifer consists of very fine to medium quartz sand, with discontinuous layers of carbonaceous clayey silt; it also contains coarse to very coarse sand and gravel , particularly in the thicker parts (Trapp, 1992)". (Masterson et al., 2015).
Confining unit	Sedimentary rock aquitard (consolidated or	"The regional confining unit overlying the Lower Chesapeake aquifer includes an unnamed confining unit in New Jersey, the Saint Marys confining unit in Delaware, Maryland, and Virginia, and the Pungo River confining unit in

Formation name	Category	Quote
	semi-consolidated rock	North Carolina. This Miocene unit primarily composes silt and clay but is diatomaceous in New Jersey and silty and shelly in Delaware, Maryland, and Virginia (Trapp, 1992)." (Masterson et al., 2015).
Middle Potomac-Raritan-Magothy aquifer	Unconsolidated aquifer	"In New Jersey, the Potomac aquifer system is subdivided into the Middle Potomac-Raritan-Magothy and Lower Potomac-Raritan-Magothy aquifers " (Masterson et al., 2015). "Unconsolidated Cretaceous-age sediments of fluvial-deltaic origin comprise an aquifer system that is the thickest, deepest, and most important source of groundwater throughout the NACP. This Potomac aquifer system (Potomac-Patuxent and Potomac-Patapsco aquifers, fig. 9) overlies basement bedrock". (Masterson et al., 2015).
Basement bedrock	Endogenous rock	"In the inner part of the crater, the Exmore clast confining unit directly overlies the altered bedrock basement and reaches its maximum thickness of more than 4,500 ft. ". (Masterson et al., 2015).

Masterson, J.P., Pope, J.P., Monti, Jack, Jr., Nardi, M.R., Finkelstein, J.S., McCoy, K.J. (2015). Hydrogeology and hydrologic conditions of the Northern Atlantic Coastal Plain aquifer system from Long Island, New York, to North Carolina (ver. 1.1, September 2015); U.S. Geological Survey Scientific Investigations Report 2013–5133, 76 p., <http://dx.doi.org/10.3133/sir20135133>

3.33 North Carolina and Virginia Coastal Plain, North Atlantic Coastal Plain

Supplementary Fig. 163. Hydrogeologic cross section. 20 equally spaced transparent pink bars overlie the cross section; each shaded bar depicts the vertical offset from the land surface to the top of the uppermost confining unit or endogenous bedrock.

Supplementary Fig. 164. Vertical variations in the prevalence of wells that have been defined as tapping an unconfined or a confined aquifer by the USGS. The smaller squares represent 10 m depth intervals from the land surface to 100 m; the larger squares represent 20 m intervals from 100 m to 300 m below the land surface.

The North Carolina and Virginia Coastal Plain subarea of the broader North Atlantic Coastal Plain is located in northeastern North Carolina and southeastern Virginia.

(i) A hydrogeologic cross section presented in Plate 9 by Winner Jr. and Coble (1989) depicts a shallow surficial aquifer underlain by a confining unit at depths of less than ~50 m below the land surface.

(ii) We analysed wells within the study area that the USGS has defined as either unconfined or confined. Most (>80%) wells at depths of 30-40 m and at depths exceeding 30 m are defined as tapping a confined aquifer.

Depth to confined conditions: 30-40 meters below land surface (based on (ii) above)

Reference: Winner Jr., M.D., Coble, R.W. (1989). Hydrogeologic framework of the North Carolina Coastal Plain aquifer system. U.S. Geological Survey Report 87-690, 167 pp. Accessed April 1, 2021 from <https://pubs.usgs.gov/of/1987/0690/report.pdf>

The table below presents a series of published quotes (see quotation marks denoting text quoted from another publication, which is cited following the quotation marks with the full reference written in full below the table). The leftmost column lists a title of a hydrogeologic formation depicted in the cross section on the previous page. The rightmost column presents a quote from a hydrogeological study (see base of table for citation). The quote has been annotated with colored text to highlight how we categorized each layer (i.e., see categories in the center column in the table). Specifically: (i) blue text highlights portions of a quote that provide insights into the degree of consolidation of the formation, (ii) red text highlights portions of a quote that categorize the formation as an aquifer or an aquitard (i.e., higher versus lower permeability in the context of local hydrogeologic formations), and (iii) green text highlights portions of a quote that provide information about the lithology of the formation.

Supplementary Table 36. Hydrostratigraphy details for the North Carolina and Virginia Coastal Plain

Formation name	Category	Quote
Surficial aquifer	Unconsolidated aquifer	"The surficial aquifer (A10) overlies all of the North Carolina Coastal Plain (fig. 1) and consists of fine sand, silt, clay, shell, and peat beds. Scattered deposits of coarser-grained sediments in the unit occur in relict beach ridges or in alluvium. " (Giese et al., 1997).
Yorktown confining unit	Sedimentary rock aquitard (consolidated or semi-consolidated rock)	"The Yorktown confining unit (CU9) overlying the Yorktown aquifer is comprised of the youngest clay beds of the Yorktown Formation in most places, but locally may include clay beds of Pleistocene or Holocene age. Its thickness averages about 25 ft. , ranging from less than 10 up to 50 ft thick. It is composed largely of clay and sandy clay that locally includes beds of fine sand or shell."
Yorktown aquifer	Sedimentary rock aquifer (consolidated or semi-consolidated rock)	"The Yorktown aquifer largely consists of fine sand, silty and clayey sand, sand with shells and shell beds, some limestone, and some coarse sand beds. " (Giese et al., 1997).
Pungo River confining unit	Sedimentary rock aquitard (consolidated or semi-consolidated rock)	"The Pungo River confining unit (CU8) is formed by the upper clay beds of the Pungo River Formation and contiguous clays of the lowermost Yorktown Formation. The confining unit ranges in thickness from less than 10 ft near the western margin to about 150 ft beneath Currituck County, with an average thickness of nearly 55 ft." (Giese et al., 1997)
Pungo River aquifer	Sedimentary rock aquifer (consolidated or semi-consolidated rock)	"The Pungo River aquifer is composed of fine to medium marine sands having considerable phosphate content." (Giese et al., 1997). "The Pungo River aquifer (A8) is thinnest near its western and northern limits, where its thickness averages about 15 ft. The aquifer dips eastward and thickens to more than 200 ft in the vicinity of the Outer Banks, where the top is deeper than 700 ft below sea level. " (Giese et al., 1997)
Castle Hayne confining unit	Sedimentary rock aquitard (consolidated or semi-consolidated rock)	"The thickness of the Castle Hayne confining unit (CU7) averages only about 10 ft; it exceeds 25 ft only in Gates County along the Virginia border, in eastern Pamlico and Carteret Counties, and in two small areas along the western limit of the Castle Hayne aquifer (A7). The confining unit is composed of beds of clay, sandy clay, and clay with sandy streaks that are part of the Pungo River Formation, the

Formation name	Category	Quote
		Yorktown Formation, or younger clays. (Giese et al., 1997)
Castle Hayne aquifer	Carbonate aquifer	"The Castle Hayne aquifer (A7) consists of limestone, sand, and minor amounts of clay deposited under marine conditions. Limestone may occur as shell limestone, dolomitic limestone, and sandy limestone ranging from loosely consolidated to hard and recrystallized." (Giese et al., 1997).
Beaufort confining unit,	Sedimentary rock aquitard (consolidated or semi-consolidated rock)	"The Beaufort confining unit (CU6) consists of the uppermost sediments of the Beaufort Formation and possibly some younger clay, silt, and sandy clay. Over most of the area, the confining unit shows a gradation from sandy clay to clay, but contains distinct clay beds interlayered with fine sand or silt." (Giese et al., 1997)
Beaufort aquifer	Sedimentary rock aquifer (consolidated or semi-consolidated rock)	"The Beaufort aquifer (A6) consists of fine to medium glauconitic sands, clayey sands, and clay beds of marine origin. Shell and limestone beds are present but are less than 6 ft thick." (Giese et al., 1997).
Black Creek confining unit	Sedimentary rock aquitard (consolidated or semi-consolidated rock)	"The Black Creek confining unit (CU4) is primarily composed of the uppermost beds of the Black Creek Formation and consists of clay, silty clay, and sandy clay." (Giese et al., 1997).
Black Creek aquifer	Sedimentary rock aquifer (consolidated or semi-consolidated rock)	"The Black Creek aquifer (A4) contains Upper Cretaceous sediments of both the Black Creek and underlying Middendorf Formations (Winner and Coble, 1989, 1996). The Black Creek Formation consists mainly of thinly laminated gray to black clay, interbedded with gray to tan sands. Outcrops also exhibit sand- or clay-dominated lenses." (Giese et al., 1997).
Upper Cape Fear confining unit	Sedimentary rock aquitard (consolidated or semi-consolidated rock)	"As described by Winner and Coble (1989), the upper Cape Fear confining unit (CU3) consists of nearly continuous clay, silty clay, and sandy clay beds belonging either to the Middendorf Formation in the Sand Hills area or to the Black Creek Formation." (Giese et al., 1997).
Upper Cape Fear aquifer	Sedimentary rock aquifer (consolidated or semi-consolidated rock)	"The sediments of the upper Cape Fear aquifer (A3), (fig. 18) are alternating beds of sand and clay." (Giese et al., 1997).
Lower Cape Fear confining unit	Sedimentary rock aquitard (consolidated or semi-consolidated rock)	"The lower Cape Fear confining unit (CU2) is composed of clay and sandy-clay beds that belong largely to the Cape Fear Formation. The average thickness of the confining unit is about 50 ft." (Giese et al., 1997)
Lower Cape Fear aquifer	Sedimentary rock aquifer (consolidated or semi-consolidated rock)	"The lower Cape Fear aquifer (A2) strikes northeast and dips southwest at a rate of 15 to 35 ft/mi. Its extent is shown in figure 20. Its thickness ranges from a few feet along its western margin to more than 400 ft in the north- eastern North Carolina Coastal Plain." (Giese et al., 1997). "The Cape Fear aquifer consists predominantly of sand, silt, and gravel separated by relatively thick silt and clay layers." (Aucott, 1996).

Formation name	Category	Quote
Lower Cretaceous confining unit	Sedimentary rock aquitard (consolidated or semi-consolidated rock)	"The Lower Cretaceous confining unit (CU1) consists of clay and sandyclay beds that belong to either sediments of Early Cretaceous or Late Cretaceous age. The thickness of the unit averages about 46 ft but is nearly 70 ft in Camden and Currituck Counties. The Lower Cretaceous aquifer and confining unit are overlain everywhere by the lower Cape Fear aquifer (A2) and underlain everywhere by crystalline basement rocks (Winner and Coble, 1989)." (Giese et al., 1991).
Lower Cretaceous aquifer	Sedimentary rock aquifer (consolidated or semi-consolidated rock)	"Various investigators have established that the updip beds of the Lower Cretaceous aquifer are largely nonmarine in origin, but the incidence of beds of marine origin increases downdip toward the coast. The non-marine beds are shales, sands, and gravel. Marine beds are chiefly limestones that may be sandy or dolomitic." (Giese et al., 1997).
Basement Rocks	Endogenous bedrock	"The Lower Cretaceous aquifer and confining unit are overlain everywhere by the lower Cape Fear aquifer (A2) and are underlain everywhere by crystalline basement rocks". (Giese et al., 1997). "The unconsolidated Coastal Plain aquifer system is underlain by crystalline basement rocks of low permeability." (Giese et al., 1991).

Kennedy, C. D., Genereux, D. P. (2007). 14iC groundwater age and the importance of chemical fluxes across aquifer boundaries in confined Cretaceous aquifers of North Carolina, USA. *Radiocarbon*, 49(3), 1181-1203.

Giese G.L., Eimers J.L., Coble R.W. (1997). Simulation of ground water flow in the Coastal Plain aquifer system of North Carolina. US Geological Survey Professional Paper 1404-M. 142 p. accessed on 3/29/2022 via <https://pubs.usgs.gov/pp/1404m/report.pdf>

Aucott, W. R. (1996). *Hydrology of the Southeastern Coastal Plain aquifer system in South Carolina and parts of Georgia and North Carolina* (No. 1410-E). US Geological Survey. <https://pubs.er.usgs.gov/publication/pp1410E>

Giese, G. L., Eimers, J. L., Coble, R. W. (1991). *Simulation of ground-water flow in the Coastal Plain aquifer system of North Carolina* (Vol. 1404). US Government Printing Office. Accessed on 3/29/2022 via <https://pubs.er.usgs.gov/publication/ofr90372>

3.34 Powder River Basin, Northern Great Plains

Supplementary Fig. 165. Hydrogeologic cross section. 20 equally spaced transparent pink bars overlie the cross section; each shaded bar depicts the vertical offset from the land surface to the top of the uppermost confining unit or endogenous bedrock.

Supplementary Fig. 166. Vertical variations in the prevalence of wells that have been defined as tapping an unconfined or a confined aquifer by the USGS. The smaller squares represent 10 m depth intervals from the land surface to 100 m; the larger squares represent 20 m intervals from 100 m to 300 m below the land surface.

The Powder River Basin is located in northeastern Wyoming and southeastern Montana.

(i) A hydrogeologic cross section presented in Fig. 4 by Long et al. (2018) depicts relatively shallow confining units in some areas, and deeper depths to the uppermost confining unit in other areas.

(ii) We analysed wells within the study area that the USGS has defined as either unconfined or confined. Most (>80%) wells at depths of 60-70 m and at depths exceeding 60 m are defined as tapping a confined aquifer.

Depth to confined conditions: 60-70 m (see (ii) above)

Reference: Long, A.J., Thamke, J.N., Davis, K.W., Bartos, T.T. (2018). Groundwater availability of the Williston Basin, United States and Canada: U.S. Geological Survey Professional Paper 1841, 42 p., Accessed February 16, 2021 from <https://doi.org/10.3133/pp1841>

The table below presents a series of published quotes (see quotation marks denoting text quoted from another publication, which is cited following the quotation marks with the full reference written in full below the table). The leftmost column lists a title of a hydrogeologic formation depicted in the cross section on the previous page. The rightmost column presents a quote from a hydrogeological study (see base of table for citation). The quote has been annotated with colored text to highlight how we categorized each layer (i.e., see categories in the center column in the table). Specifically: (i) blue text highlights portions of a quote that provide insights into the degree of consolidation of the formation, (ii) red text highlights portions of a quote that categorize the formation as an aquifer or an aquitard (i.e., higher versus lower permeability in the context of local hydrogeologic formations), and (iii) green text highlights portions of a quote that provide information about the lithology of the formation.

Supplementary Table 37. Hydrostratigraphy details for the Powder River Basin

Formation name	Category	Quote
Upper Fort Union aquifer	Clastic sedimentary rock aquifer (consolidated or semi-consolidated rock)	"The upper Fort Union aquifer is as thick as 1,920 ft and composed of crossbedded light-yellow to light-yellow-gray sandstone, sandy mudstone, gray shale, carbonaceous shale, and thick coal beds and associated clinker deposits (permeable rocks created by the natural burning of coal beds)." (Long et al., 2018).
Middle Fort Union hydrogeologic unit (in some areas can be aquifer)	Sedimentary rock aquitard (consolidated or semi-consolidated rock)	" Composed of as much as 520 ft of thickness of alternating beds of sandstone, siltstone, mudstone, claystone, and lignite, rocks in the middle Fort Union hydrogeologic unit generally are finer-grained and darker-colored than the overlying upper Fort Union aquifer and underlying lower Fort Union aquifer. Because of spatially variable lithology, the middle Fort Union hydrogeologic unit may act as a confining unit in some areas and as an aquifer in other areas." (Long et al., 2018).
Lower Fort Union aquifer	Clastic sedimentary rock aquifer (consolidated or semi-consolidated rock)	"The lower Fort Union aquifer is composed of as much as 670 ft of thickness of yellow-weathering sandstones and light-gray-weathering sandy mudstones interfingering with alternating brown and gray beds of sandstone, siltstone, claystone, mudstone, and lignite deposited in continental and marine environments. " (Long et al., 2018).
Upper Hell Creek hydrogeologic unit (in some areas can be aquifer)	Sedimentary rock aquitard (consolidated or semi-consolidated rock)	"The upper Hell Creek hydrogeologic unit is composed of as much as 740 ft of thickness of alternating layers of gray and brown mudstone, siltstone, sandstone, and sparse lignite beds deposited by meandering streams with point bars and channel plugs. Because of spatial variability, this lithology may act as a confining unit in some areas and as an aquifer in other areas." (Long et al., 2018).
Lower Hell Creek aquifer	Clastic sedimentary rock aquifer (consolidated or semi-consolidated rock)	"The general lithology of the lower Hell Creek aquifer is similar to the upper Hell Creek hydrogeologic unit, except that the latter has a smaller percentage of sandstone. " (Long et al., 2018).
Fox Hills aquifer	Clastic sedimentary rock aquifer (consolidated or semi-consolidated rock)	"The Fox Hills aquifer is the most areally extensive of the units, with as much as 420 ft of thickness of interbedded sandstone, siltstone, and mudstone. " (Long et al., 2018).
Basal confining unit	Sedimentary rock aquitard (consolidated or	"In the Williston Basin, these three aquifer systems are as deep as 2,850 ft below land surface and overlie 800–2,000 ft of relatively impermeable Upper Cretaceous marine shale

Formation name	Category	Quote
	semi-consolidated rock	that serves as a basal confining unit that impedes groundwater flow (Anna, 1986; Downey, 1986; Downey and Dinwiddie, 1988; Thamke and others, 2014)." (Long et al., 2018).

Long, A.J., Thamke, J.N., Davis, K.W., Bartos, T.T. (2018). Groundwater availability of the Williston Basin, United States and Canada: U.S. Geological Survey Professional Paper 1841, 42 p., <https://doi.org/10.3133/pp1841>

3.35 Williston Basin, Northern Great Plains

Supplementary Fig. 167. Hydrogeologic cross section. 20 equally spaced transparent pink bars overlie the cross section; each shaded bar depicts the vertical offset from the land surface to the top of the uppermost confining unit or endogenous bedrock.

Supplementary Fig. 168. Vertical variations in the prevalence of wells that have been defined as tapping an unconfined or a confined aquifer by the USGS. The smaller squares represent 10 m depth intervals from the land surface to 100 m; the larger squares represent 20 m intervals from 100 m to 300 m below the land surface.

The Williston Basin is located in eastern Montana, western North Dakota, and northwestern South Dakota.

(i) A hydrogeologic cross section presented in Fig. 4 by Long et al. (2018) depicts complex layered sedimentary sequences, with relatively deep depths to the uppermost confining unit in the northeast portion of the hydrogeologic study area.

(ii) We analysed wells within the study area that the USGS has defined as either unconfined or confined. Most (>80%) wells at depths of 140-160 m and at depths exceeding 140 m are defined as tapping a confined aquifer.

Depth to confined conditions: 140-160 m (see (ii) above)

Reference: Long, A.J., Thamke, J.N., Davis, K.W., Bartos, T.T. (2018). Groundwater availability of the Williston Basin, United States and Canada: U.S. Geological Survey Professional Paper 1841, 42 p., Accessed February 16, 2021 from <https://doi.org/10.3133/pp1841>

The table below presents a series of published quotes (see quotation marks denoting text quoted from another publication, which is cited following the quotation marks with the full reference written in full below the table). The leftmost column lists a title of a hydrogeologic formation depicted in the cross section on the previous page. The rightmost column presents a quote from a hydrogeological study (see base of table for citation). The quote has been annotated with colored text to highlight how we categorized each layer (i.e., see categories in the center column in the table). Specifically: (i) blue text highlights portions of a quote that provide insights into the degree of consolidation of the formation, (ii) red text highlights portions of a quote that categorize the formation as an aquifer or an aquitard (i.e., higher versus lower permeability in the context of local hydrogeologic formations), and (iii) green text highlights portions of a quote that provide information about the lithology of the formation.

Supplementary Table 38. Hydrostratigraphy details for the Williston Basin

Formation name	Category	Quote
Glacial aquifer system	Unconsolidated aquifer	"The glacial aquifer system consists of Quaternary-age unconsolidated till, silt, clay, outwash sand and gravel, and occasional cobbles and boulders. " (Long et al., 2018).
Upper Fort Union aquifer	Clastic sedimentary rock aquifer (consolidated or semi-consolidated rock)	"The upper Fort Union aquifer is as thick as 1,920 ft and composed of crossbedded light-yellow to light-yellow-gray sandstone, sandy mudstone, gray shale, carbonaceous shale, and thick coal beds and associated clinker deposits (permeable rocks created by the natural burning of coal beds)." (Long et al., 2018).
Middle Fort Union hydrogeologic unit (in some areas can be aquifer)	Sedimentary rock aquitard (consolidated or semi-consolidated rock)	" Composed of as much as 520 ft of thickness of alternating beds of sandstone, siltstone, mudstone, claystone, and lignite, rocks in the middle Fort Union hydrogeologic unit generally are finer-grained and darker-colored than the overlying upper Fort Union aquifer and underlying lower Fort Union aquifer. Because of spatially variable lithology, the middle Fort Union hydrogeologic unit may act as a confining unit in some areas and as an aquifer in other areas. " (Long et al., 2018).
Lower Fort Union aquifer	Clastic sedimentary rock aquifer (consolidated or semi-consolidated rock)	"The lower Fort Union aquifer is composed of as much as 670 ft of thickness of yellow-weathering sandstones and light-gray-weathering sandy mudstones interfingering with alternating brown and gray beds of sandstone, siltstone, claystone, mudstone, and lignite deposited in continental and marine environments. " (Long et al., 2018).
Upper Hell Creek hydrogeologic unit (in some areas can be aquifer)	Sedimentary rock aquitard (consolidated or semi-consolidated rock)	"The upper Hell Creek hydrogeologic unit is composed of as much as 740 ft of thickness of alternating layers of gray and brown mudstone, siltstone, sandstone, and sparse lignite beds deposited by meandering streams with point bars and channel plugs. Because of spatial variability, this lithology may act as a confining unit in some areas and as an aquifer in other areas." (Long et al., 2018).
Lower Hell Creek aquifer	Clastic sedimentary rock aquifer (consolidated or semi-consolidated rock)	"The general lithology of the lower Hell Creek aquifer is similar to the upper Hell Creek hydrogeologic unit, except that the latter has a smaller percentage of sandstone. " (Long et al., 2018).

Formation name	Category	Quote
Fox Hills aquifer	Clastic sedimentary rock aquifer (consolidated or semi-consolidated rock)	"The Fox Hills aquifer is the most areally extensive of the units, with as much as 420 ft of thickness of interbedded sandstone, siltstone, and mudstone." (Long et al., 2018).
Basal confining unit	Sedimentary rock aquitard (consolidated or semi-consolidated rock)	"In the Williston Basin, these three aquifer systems are as deep as 2,850 ft below land surface and overlie 800–2,000 ft of relatively impermeable Upper Cretaceous marine shale that serves as a basal confining unit that impedes groundwater flow (Anna, 1986; Downey, 1986; Downey and Dinwiddie, 1988; Thamke and others, 2014)." (Long et al., 2018).

Long, A.J., Thamke, J.N., Davis, K.W., Bartos, T.T. (2018). Groundwater availability of the Williston Basin, United States and Canada: U.S. Geological Survey Professional Paper 1841, 42 p., <https://doi.org/10.3133/pp1841>

3.36 Eastern Silurian-Devonian Aquifers, Northern Midwest Aquifer System

Supplementary Fig. 169. Hydrogeologic cross section. 20 equally spaced transparent pink bars overlie the cross section; each shaded bar depicts the vertical offset from the land surface to the top of the uppermost confining unit or endogenous bedrock.

Supplementary Fig. 170. Vertical variations in the prevalence of wells that have been defined as tapping an unconfined or a confined aquifer by the USGS. The smaller squares represent 10 m depth intervals from the land surface to 100 m; the larger squares represent 20 m intervals from 100 m to 300 m below the land surface.

The Eastern Silurian-Devonian Aquifers subarea of the broader Northern Midwest Aquifer System is located in eastern Illinois, west of Lake Michigan.

(i) A hydrogeologic cross section presented in Fig. 20 by Young (1992) shows a drift aquifer atop the Silurian-Devonian carbonates, underlain by the Maquoketa confining unit.

(ii) We analysed wells within the study area that the USGS has defined as either unconfined or confined. Most (>80%) wells at depths of 30-40 m and at depths exceeding 30 m are defined as tapping a confined aquifer.

Depth to confined conditions: 30-40 m (see (ii) above)

Reference: Young, H.L. (1992). Hydrogeology of the Cambrian-Ordovician Aquifer System in the Northern Midwest, United States. U.S. Geological Survey Professional Paper 1405-B, 108 pp. Accessed April 12, 2021 from <https://pubs.usgs.gov/pp/1405b/report.pdf>

The table below presents a series of published quotes (see quotation marks denoting text quoted from another publication, which is cited following the quotation marks with the full reference written in full below the table). The leftmost column lists a title of a hydrogeologic formation depicted in the cross section on the previous page. The rightmost column presents a quote from a hydrogeological study (see base of table for citation). The quote has been annotated with colored text to highlight how we categorized each layer (i.e., see categories in the center column in the table). Specifically: (i) blue text highlights portions of a quote that provide insights into the degree of consolidation of the formation, (ii) red text highlights portions of a quote that categorize the formation as an aquifer or an aquitard (i.e., higher versus lower permeability in the context of local hydrogeologic formations), and (iii) green text highlights portions of a quote that provide information about the lithology of the formation.

Supplementary Table 39. Hydrostratigraphy details for the Eastern Silurian Devonian

Formation name	Category	Quote
Drift aquifer	Unconsolidated aquifer	" Aquifer layer " Young, H.L. (1992) "Because glacial drift is almost universally present in the study area and is the shallowest permeable rock material, it provides a ready source of ground water. Ground-water availability from the drift is directly proportional to the amount of permeable, well-sorted sand and gravel within the drift . The large variety of sediment types in the drift have a wide range of sorting and depositional form" Young, H.L. (1992). " Sand and gravel is less abundant, present as discontinuous lenses within or beneath moraine or as more extensive outwash and ice-contact deposits. " Young (1992)
Silurian-Devonian aquifer	Carbonate aquifer	" Aquifer layer " Young, H.L. (1992) " Dolomite and limestone constitute most of the Silurian through Middle Devonian rocks in the study area, which are collectively termed the Silurian-Devonian aquifer in this study. The rock is relatively fine grained and dense, and its permeability is primarily dependent on the extent and degree of intersection of fractures and joints within it and on the subsequent solutional enlargement of these openings by weathering action and ground-water movement." Young (1992)
Maquoketa confining unit	Sedimentary rock aquitard (consolidated or semi-consolidated rock)	" Confining layer " Young, H.L. (1992) "This confining unit consists of the Maquoketa Shale, the Galena Dolomite, and the Decorah, Platteville, and Glenwood Formations or equivalents. " Young, H.L. (1992)
St. Peter-Prairie du Chien-Jordan aquifer	Clastic sedimentary rock aquifer (consolidated or semi-consolidated rock)	" Aquifer layer " Young, H.L. (1992) "Although this multiunit aquifer may be the least uniform of the bedrock aquifer units in the northern Midwest (fig. 18), it is a major source of ground water in Iowa and Minnesota . In these States, the Jordan Sandstone and all or parts of the overlying Prairie du Chien Group are in direct hydraulic connection, resulting in a highly productive aquifer." Young (1992)
St. Lawrence-Franconia confining unit	Sedimentary rock aquitard (consolidated or semi-consolidated rock)	" Confining layer " Young (1992) "The St. Lawrence and Franconia Formations form an important regional confining unit over the Ironton-Galesville aquifer. Although these formations are dominantly sandstone in the northern part of the area, they are very silty and shaly, fine grained, poorly sorted, and dolomitic . Thus, the units are anisotropic and restrict vertical movement of ground water. " Young (1992)

Formation name	Category	Quote
Ironton-Galesville aquifer	Clastic sedimentary rock aquifer (consolidated or semi-consolidated rock)	" Aquifer layer " Young, H.L. (1992) "The Ironton and Galesville Sandstones form the most important aquifer of the Cambrian-Ordovician aquifer system in the east-central part of the study area, although they generally are not the only rock units open to deep wells." Young, H.L. (1992). "The aquifer terminates to the west, south, and east (fig. 10) as the sandstones grade into carbonate rocks, primarily dolomite. " Young (1992)
Eau Claire confining unit	Sedimentary rock aquitard (consolidated or semi-consolidated rock)	" Confining layer " Young (1992) "The Eau Claire Formation and its partial equivalent to the southwest, the Bonneterre Formation, form an extensive confining unit above the Mount Simon aquifer. The effectiveness of the Eau Claire as a confining unit depends on the relative abundance of shale, siltstone, dolomite, and sandstone in the formation. " Young (1992)
Mount Simon aquifer	Clastic sedimentary rock aquifer (consolidated or semi-consolidated rock)	" Aquifer layer " Young (1992) "The lowermost aquifer of the Cambrian-Ordovician aquifer system is composed primarily of the Mount Simon Sandstone and equivalent strata, the Lamotte Sandstone, in Missouri. In Minnesota, the Mount Simon is underlain by Precambrian sedimentary rocks-the Hinckley Sandstone and the older Fond du Lac Formation." Young (1992)
Precambrian crystalline rocks (confining unit)	Endogenous bedrock	Young, 1992 Figure 4, System: " Precambrian ": major rock type: " Igneous and metamorphic crystalline rocks ". " The dense crystalline rocks of the Precambrian basement beneath the Cambrian-Ordovician aquifer system are a very effective confining unit whose upper surface marks the lower limit of the Cambrian-Ordovician aquifer system." (Young, 1992)

Young, H.L. (1992). Hydrogeology of the Cambrian-Ordovician Aquifer System in the Northern Midwest, United States. U.S. Geological Survey Professional Paper 1405-B, 108 pp. Accessed April 12, 2021 from <https://pubs.usgs.gov/pp/1405b/report.pdf>

3.37 Mississippian-Silurian-Devonian Carbonates, Northern Midwest Aquifer System

Supplementary Fig. 171. Hydrogeologic cross section. 20 equally spaced transparent pink bars overlie the cross section; each shaded bar depicts the vertical offset from the land surface to the top of the uppermost confining unit or endogenous bedrock.

Supplementary Fig. 172. Vertical variations in the prevalence of wells that have been defined as tapping an unconfined or a confined aquifer by the USGS. The smaller squares represent 10 m depth intervals from the land surface to 100 m; the larger squares represent 20 m intervals from 100 m to 300 m below the land surface.

The Mississippian-Silurian-Devonian Carbonates subarea of the broader Northern Midwest Aquifer System is located in eastern Illinois, west of Lake Michigan.

(i) A hydrogeologic cross section presented in Fig. 20 by Young (1992) shows Pleistocene deposits overlying Mississippian- to Silurian-aged carbonate rocks atop a series of confining units.

(ii) We analysed wells within the study area that the USGS has defined as either unconfined or confined. Most (>80%) wells at depths of 60-70 m and at depths exceeding 60 m are defined as tapping a confined aquifer.

Depth to confined conditions: 60-70 m (see (ii) above)

Reference: Young, H.L. (1992). Hydrogeology of the Cambrian-Ordovician Aquifer System in the Northern Midwest, United States. U.S. Geological Survey Professional Paper 1405-B, 108 pp. Accessed April 12, 2021 from <https://pubs.usgs.gov/pp/1405b/report.pdf>

The table below presents a series of published quotes (see quotation marks denoting text quoted from another publication, which is cited following the quotation marks with the full reference written in full below the table). The leftmost column lists a title of a hydrogeologic formation depicted in the cross section on the previous page. The rightmost column presents a quote from a hydrogeological study (see base of table for citation). The quote has been annotated with colored text to highlight how we categorized each layer (i.e., see categories in the center column in the table). Specifically: (i) blue text highlights portions of a quote that provide insights into the degree of consolidation of the formation, (ii) red text highlights portions of a quote that categorize the formation as an aquifer or an aquitard (i.e., higher versus lower permeability in the context of local hydrogeologic formations), and (iii) green text highlights portions of a quote that provide information about the lithology of the formation.

Supplementary Table 40. Hydrostratigraphy details for the Mississippian-Silurian-Devonian

Formation name	Category	Quote
Q – Pleistocene deposits	Unconsolidated aquifer	“unconsolidated rocks (glacial drift and alluvial deposits) and bedrock aquifers that store large quantities of ground water.” (Young, 1992). Young, 1992 Figure 4. “Unconsolidated deposits of clay, silt, sand, gravel, and boulders; degree of sorting variable”
IP – Pennsylvanian rocks	Sedimentary rock aquitard (consolidated or semi-consolidated rock)	Figure 4 of (Young, 1992) under major rock units “Limestone and shale” in Iowa and in Northern Illinois “Limestone, sandstone, siltstone, limestone, coal and clay” . Figure 18 of (Young, 1992). “Confining layer” or “Confining beds”
M – Mississippian rocks (It is absent in Wisconsin, and aquifer in Iowa and northern Illinois)	Carbonate aquifer	Young, 1992 Figure 4, System: “Mississippian”; major rock type in Northern Missouri: “Limestone, sandstone, siltstone, and shale” . “The Kinderhookian Series is mainly shale and siltstone in Illinois and northern Missouri, but the upper part contains some limestone . The limestone increases in abundance northwestward.” (Young, 1992). “The uppermost series in the Mississippian, the Chesterian Series, is present only in the southern parts of Missouri, Illinois, and Indiana. It consists of repetitive beds of sandstone, shale, and limestone that are classified into 20 formations in southern Illinois.” (Young, 1992). Young, 1992 Figure 18. Hydrogeologic unit: Pennsylvanian-Mississippian-Devonian confining unit , Layer number in digital flow model: “Confining layer” .
D – Devonian rocks (In Northern Illinois: it is part of the “Upper Confining Unit”.	Carbonate aquifer	Young, 1992 Figure 4, System: “Devonian”; major rock type in Iowa: “Limestone and shale” , Northern Illinois: “Dolomite and limestone” . Young, 1992 Figure 18, Iowa: “Silurian-Devonian aquifer” , and Northern Illinois: “Devonian or Upper confining bed” .
S – Silurian rocks	Carbonate aquifer	Young, 1992 Figure 4, System: “Silurian” major rocks in Iowa and Northern Illinois: “Dolomite” and “Dolomite and limestone” respectively. “Rocks of Middle Devonian through Silurian age are mainly dolomite and limestone and are termed the Silurian-Devonian aquifer .” (Young, 1992).
Om – Maquoketa Shale	Sedimentary rock aquitard (consolidated or semi-consolidated rock)	Young, 1992 Figure 18, Rock stratigraphic unit: “Maquoketa shale” . Wisconsin, Iowa and Northern Illinois: “Confining bed” . “This confining unit consists of the Maquoketa Shale” (Young, 1992).

Formation name	Category	Quote
Okd – Kimmswick and Decorah Formations (Kimmswick can be aquifer in other states)	Sedimentary rock aquitard (consolidated or semi-consolidated rock)	“The Galena Dolomite of Wisconsin is the rock stratigraphic equivalent of the Kimmswick Formation of Missouri.” (Young, 1992). “The Galena Dolomite is primarily carbonate, either dolomitic limestone or interbedded limestone and dolomite.” (Young, 1992). Young, 1992 Figure 18, Rock stratigraphic unit: “Maquoketa shale”. Wisconsin, Iowa and Northern Illinois: “Confining bed”. “This confining unit consists of the Maquoketa Shale” (Young, 1992). Young, 1992 Figure 18 “confining unit” in Iowa and “Galena-Platteville unit” in Northern Illinois.
Og – Galena Group – Consists of Dubuque, Wise Lake, and Dunleith Formations and Decorah Formation in Iowa or Guttenberg and Spechts Ferry Formations in Illinois.	Sedimentary rock aquitard (consolidated or semi-consolidated rock)	“The Galena Dolomite is primarily carbonate, either dolomitic limestone or interbedded limestone and dolomite.” (Young, 1992). Young, 1992 Figure 18, Rock stratigraphic unit: “Maquoketa shale”. Wisconsin, Iowa and Northern Illinois: “Confining bed”. “This confining unit consists of the Maquoketa Shale” (Young, 1992). Young, 1992 Figure 18 “confining unit” in Iowa and “Galena-Platteville unit” in Northern Illinois.
Op – Platteville Group – Consists of Quimbys Mill, Nachusa, Grand Detour, Mifflin, and Pecatonica Formations in Illinois or Platteville Formation in Iowa	Sedimentary rock aquitard (consolidated or semi-consolidated rock)	“The Maquoketa Shale is its main component, but where the Maquoketa overlies dolomite and shale of the Galena Dolomite and the Decorah, Platteville, and Glenwood Formations, these units also are an important part of the confining unit.” (Young, 1992). Platteville Formation: The lowermost member of the Platteville, the Pecatonica Member, is mainly a yellowish- to grayish-brown, fine- to medium-grained, medium-bedded to massive dolomite that commonly contains some sand near its base.” (Young, 1992). “This confining unit consists of the Maquoketa Shale, the Galena Dolomite, and the Decorah, Platteville, and Glenwood Formations or equivalents.” (Young, 1992).
Oqp – Galena Dolomite, Decorah and Platteville Formations	Sedimentary rock aquitard (consolidated or semi-consolidated rock)	“The Galena Dolomite is primarily carbonate, either dolomitic limestone or interbedded limestone and dolomite.” (Young, 1992). Young, 1992 Figure 18, Rock stratigraphic unit: “Maquoketa shale”. Wisconsin, Iowa and Northern Illinois: “Confining bed”. “This confining unit consists of the Maquoketa Shale” (Young, 1992). Young, 1992 Figure 18 “confining unit” in Iowa and “Galena-Platteville unit” in Northern Illinois.
Osp – St. Peter Sandstone	Sedimentary rock aquifer (consolidated or semi-consolidated rock)	Young, 1992 Figure 18, Iowa: “St. Peter Ss”, rock stratigraphic unit and “water bearing”. “A prominent exception to the carbonate facies is the St. Peter Sandstone—a very well sorted, pure quartzose sandstone that is very extensive and uniform throughout the northern Midwest.” (Young, 1992).
Oa – Ansell Group – Consists of Glenwood Formation, St. Peter Sandstone	Sedimentary rock aquifer (consolidated or semi-consolidated rock)	Young, 1992 Figure 18, Northern Illinois: “St. Peter Ss”, rock stratigraphic unit and “Ansell aquifer”. “A prominent exception to the carbonate facies is the St. Peter Sandstone—a very well sorted, pure quartzose sandstone that is very extensive and uniform throughout the northern Midwest.” (Young, 1992).

Formation name	Category	Quote
in Iowa, Illinois, and Wisconsin		
Ocj – Cotter and Jefferson City Dolomites	Carbonate aquifer	"In northern Missouri, the Jefferson City and Cotter Dolomites , equivalent to the Willow River Member, consist primarily of fine- to medium-grained dolomite with variable amounts of chert and thin beds of shale and fine-grained sandstone. " (Young, 1992). "The upper most Prairie du Chien, the Willow River Member of the Shakopee Formation , and the basal beds of the St. Peter locally confine the Prairie du Chien and Jordan. Equivalent rocks to the Prairie du Chien in northern Missouri, the Roubidoux Formation and Gasconade Dolomite, are mainly carbonate, but they also contain enough sandstone to make them productive and important aquifers." (Young, 1992). Young, 1992 Figure 18, both in Iowa and Northern Illinois this unit is part of " Cambrian-Ordovician aquifer "
Or – Roubidoux Formation	Carbonate aquifer	Young, 1992 Figure 18, Northern Missouri: "Roubidoux Fm., Gasconade Dol., Eminence Dol., Potosi Dol. (good aquifer)". "The Roubidoux Formation in Missouri, equivalent to the New Richmond Sandstone Member, consists of fine- to medium-grained, white, partly dolomitic, quartzose sandstone and fine-grained dolomite with beds of chert. The Roubidoux ranges in thickness from 100 to 250 ft and is thinnest in northeastern Missouri. " (Young, 1992).
Oqd – Gasconade Dolomite	Carbonate aquifer	Young, 1992 Figure 18, Northern Missouri: "Roubidoux Fm., Gasconade Dol. , Eminence Dol., Potosi Dol. (good aquifer)". "The Gasconade Dolomite in Missouri, equivalent to the Oneota, is coarse grained and very cherty in the lower part and fine grained and much less cherty in the upper part. " (Young, 1992).
Opc – Prairie du Chien Group – Consists of Shakopee, New Richmond, and Oneota rock units of various rank in Wisconsin, Iowa, and Illinois, plus the Gunter Sandstone in Illinois	Sedimentary rock aquifer (consolidated or semi-consolidated rock)	"The upper most Prairie du Chien, the Willow River Member of the Shakopee Formation , and the basal beds of the St. Peter locally confine the Prairie du Chien and Jordan. Equivalent rocks to the Prairie du Chien in northern Missouri, the Roubidoux Formation and Gasconade Dolomite, are mainly carbonate, but they also contain enough sandstone to make them productive and important aquifers." (Young, 1992). Young, 1992 Figure 18, both in Iowa and Northern Illinois this unit is part of " Cambrian-Ordovician aquifer "
Os – Shakopee Dolomite (portion of it might be aquitard in Northern Illinois, Young, 1992 of Figure 18).	Carbonate aquifer	"The major part of the Shakopee Formation consists of the Willow River Member, generally a very fine grained to fine-grained, sandy, light gray to tan or buff dolomite. " (Young, 1992). Young, 1992 Figure 18, in Iowa "Cambrian-Ordovician aquifer system" and in Northern Illinois". "The New Richmond Sandstone Member of the Shakopee Formation of the Prairie du Chien Group is somewhat similar to the Jordan, but it is thinner and more variable. In some areas, the New Richmond probably contributes significantly to the yield of wells. " (Young, 1992).

Formation name	Category	Quote
Onr – New Richmond Sandstone or Member	Sedimentary rock aquifer (consolidated or semi-consolidated rock)	"The New Richmond Sandstone Member generally consists of three lithologies: a lower fine- to medium grained, white to light gray or tan, quartzose sandstone; a transitional fine-grained, sandy dolomite; and an upper very thin (1 ft or less), blue to green shale or sandy shale (M.E. Ostrom: Wisconsin Geological and Natural History Survey, written commun., 1987)." (Young, 1992). "In some areas, the New Richmond probably contributes significantly to the yield of wells. " (Young, 1992).
Oo – Oneota Dolomite	Carbonate aquifer	"The Oneota Dolomite is fine to coarse grained, tan to light gray, with variable amounts of chert and sand. It is typically sandy in the lower part in Wisconsin, but in Illinois the lower part is very cherty with very little sand." (Young, 1992). Young, 1992 Figure 18, both in Iowa and Northern Illinois this unit is part of " Cambrian-Ordovician aquifer ".
Ee – Eminence Dolomite	Sedimentary rock aquitard (consolidated or semi-consolidated rock)	Young, 1992 Figure 18, Northern Illinois: " Eminence Dol., Potosi Dol " are part of " Middle confining units ". "In northern Missouri-the Roubidoux Formation and the Gasconade and Eminence Dolomites-are mainly carbonate rocks that are somewhat permeable and contain some sandstone. " (Young, 1992).
Ep – Potosi Dolomite	Sedimentary rock aquitard (consolidated or semi-consolidated rock)	Young, 1992 Figure 18, Northern Illinois: "Eminence Dol., Potosi Dol " are part of " Middle confining units ". " The most productive parts of the aquifer system in this area are the Roubidoux Formation and Gasconade, Eminence, and Potosi Dolomites (fig. 18)" (Young, 1992).
Esl – St. Lawrence Formation	Sedimentary rock aquitard (consolidated or semi-consolidated rock)	" Middle Confining unit " (Young, 1992) in Northern Illinois. " Confining bed " (Young, 1992) in Iowa. " Potosi Dolomite (equivalent to St. Lawrence Formation elsewhere)" (Young, 1992). "Both are fine- to medium-crystalline dolomites that contain algal material. " (Young, 1992).
Edd – Derby-Doerun Dolomite	Sedimentary rock aquitard (consolidated or semi-consolidated rock)	Young, 1992 Figure 18, Northern Missouri: " Derby-Doerun Dol. (poor aquifer) ". "In Missouri, Imes (1985) includes the Potosi and Derby-Doerun in his Cambrian Ordovician aquifer (fig. 18), although the Derby-Doerun is not considered to be very productive. " (Young, 1992). "In the southern and eastern parts of the area, the Potosi and Derby-Doerun Dolomites and the upper part of the Davis Formation are the equivalent rocks (mainly carbonate rocks). " (Young, 1992).
Ef – Franconia Formation	Sedimentary rock aquitard (consolidated or semi-consolidated rock)	" Middle Confining unit " (Young, 1992). "St. Lawrence- Franconia confining unit -The two formations that compose this unit generally are very silty and shaly, fine grained, poorly sorted, dolomitic sandstones. " (Young, 1992).
Etc – Tunnel City Group – Consists of Lone Rock and Mazomanie Formations in Wisconsin and Iowa	Sedimentary rock aquitard (consolidated or semi-consolidated rock)	Young 1992, Figure 6: Geologic unit: "Tunnel City group", the rock type is " Shaly sandstone and/or shale ". " The Franconia, whose Mazomanie Member is coarse grained, has been included with the Ironton-Galesville aquifer in Minnesota by some investigators (Kanivetsky, 1978; Adolphson and others, 1981). However, natural-gamma well logs show the Franconia there to be primarily a confining unit (Woodward, 1986)." (Young, 1992).

Formation name	Category	Quote
Ed – Davis Formation	Sedimentary rock aquitard (consolidated or semi-consolidated rock)	Young, 1992 Figure 18, Northern Missouri: " The Davis Formation, Lower confining bed ". " The Davis Formation in Missouri consists of silty to sandy shale and limestone or dolomite, with beds of flat-pebble conglomerate. " (Young, 1992). "The Davis also is a dolomite but contains as much as 50 percent shale and silt. " (Young, 1992)
Eb – Bonneterre Formation	Sedimentary rock aquitard (consolidated or semi-consolidated rock)	Young, 1992 Figure 18, Northern Missouri: "Bonneterre Fm. (little information)". " the Bonneterre Formation, form an extensive confining unit above the Mount Simon aquifer. " (Young, 1992). "The Bonneterre generally is a fine- to coarse-grained, shaly, dolomitized calcarenite with much oolitic and stromatolitic material ". "" (Young, 1992).
EI – Lamotte Sandstone	Sedimentary rock aquifer (consolidated or semi-consolidated rock)	Young, 1992 Figure 18, Northern Missouri: " Lamotte Ss. (probable aquifer) ". " The lowermost aquifer of the Cambrian-Ordovician aquifer system is composed primarily of the Mount Simon Sandstone and equivalent strata, the Lamotte Sandstone, in Missouri. " (Young, 1992). "It consists mainly of the Mount Simon Sandstone in the north and its equivalent, the Lamotte Sandstone, in northern Missouri " (Young, 1992).
EIq – Ironton Galesville Sandstones	Sedimentary rock aquifer (consolidated or semi-consolidated rock)	"The Ironton and Galesville Sandstones form the most important aquifer of the Cambrian-Ordovician aquifer system in the east-central part of the study area" (Young, 1992). "It consists of the moderately sorted, quartzose Ironton and Galesville Sandstones and generally is 50 to 150 ft thick." (Young, 1992).
Eec – Eau Claire Formation	Sedimentary rock aquitard (consolidated or semi-consolidated rock)	Young, 1992 Figure 18, Northern Missouri: " Eau Claire confining unit ". " The Eau Claire Formation and its partial equivalent to the southwest, the Bonneterre Formation, form an extensive confining unit above the Mount Simon aquifer. " (Young, 1992). "The effectiveness of the Eau Claire as a confining unit depends on the relative abundance of shale, siltstone, dolomite, and sandstone in the formation. " (Young, 1992).
Ems – Mount Simon Sandstone	Sedimentary rock aquifer (consolidated or semi-consolidated rock) Checked	Young, 1992 Figure 18, Hydrogeologic unit: " Mount Simon aquifer ". "The Mount Simon Sandstone generally is a medium to coarse-grained, poorly to moderately sorted, sometimes pebbly, white to gray, quartzose sandstone. " (Young, 1992).
pE – Precambrian rocks, undivided	Endogenous bedrock	Young, 1992 Figure 4, System: " Precambrian "; major rock type in Northern Missouri: " Igneous and metamorphic crystalline rocks ". " The dense crystalline rocks of the Precambrian basement beneath the Cambrian-Ordovician aquifer system are a very effective confining unit whose upper surface marks the lower limit of the Cambrian-Ordovician aquifer system." (Young, 1992).

Young, H.L. (1992). Hydrogeology of the Cambrian-Ordovician Aquifer System in the Northern Midwest, United States. U.S. Geological Survey Professional Paper 1405-B, 108 pp. Accessed June 14, 2022 via <https://pubs.er.usgs.gov/publication/pp1405B>

3.38 Northeast Missouri Carbonates, Northern Midwest Aquifer System

Supplementary Fig. 173. Hydrogeologic cross section. 20 equally spaced transparent pink bars overlie the cross section; each shaded bar depicts the vertical offset from the land surface to the top of the uppermost confining unit or endogenous bedrock.

Supplementary Fig. 174. Vertical variations in the prevalence of wells that have been defined as tapping an unconfined or a confined aquifer by the USGS. The smaller squares represent 10 m depth intervals from the land surface to 100 m; the larger squares represent 20 m intervals from 100 m to 300 m below the land surface.

The Northeast Missouri Carbonates subarea is located in northeastern Missouri.

(i) A hydrogeologic cross section presented in Plate 1 by Young (1992) depicts layered clastic and carbonate sedimentary units with relatively shallow confining units in most of the area.

(ii) We analysed wells within the study area that the USGS has defined as either unconfined or confined. Most (>80%) wells at depths of 50-60 m and at depths exceeding 50 m are defined as tapping a confined aquifer.

Depth to confined conditions: 50-60 m (see (ii) above)

Reference: Young, H.L. (1992). Hydrogeology of the Cambrian-Ordovician Aquifer System in the Northern Midwest, United States. U.S. Geological Survey Professional Paper 1405-B, 108 pp. Accessed April 12, 2021 from <https://pubs.usgs.gov/pp/1405b/report.pdf> (for Plate 1 see <https://pubs.usgs.gov/pp/1405b/plate-1.pdf>)

The table below presents a series of published quotes (see quotation marks denoting text quoted from another publication, which is cited following the quotation marks with the full reference written in full below the table). The leftmost column lists a title of a hydrogeologic formation depicted in the cross section on the previous page. The rightmost column presents a quote from a hydrogeological study (see base of table for citation). The quote has been annotated with colored text to highlight how we categorized each layer (i.e., see categories in the center column in the table). Specifically: (i) blue text highlights portions of a quote that provide insights into the degree of consolidation of the formation, (ii) red text highlights portions of a quote that categorize the formation as an aquifer or an aquitard (i.e., higher versus lower permeability in the context of local hydrogeologic formations), and (iii) green text highlights portions of a quote that provide information about the lithology of the formation.

Supplementary Table 41. Hydrostratigraphy details for the Northeast Missouri Carbonate

Formation name	Category	Quote
Q – Pleistocene deposits	Unconsolidated aquifer	" unconsolidated rocks (glacial drift and alluvial deposits) and bedrock aquifers that store large quantities of ground water. " (Young, 1992). Young, 1992 Figure 4. " Unconsolidated deposits of clay, silt, sand, gravel, and boulders; degree of sorting variable "
M – Mississippian rocks (Keokuk-Burlington is an aquifer and St. Louis-Salem are poor aquifer)	Sedimentary rock aquitard (consolidated or semi-consolidated rock)	Young, 1992 Figure 4, System: "Mississippian"; major rock type in Northern Missouri: " Limestone, sandstone, siltstone, and shale ". "The Kinderhookian Series is mainly shale and siltstone in Illinois and northern Missouri, but the upper part contains some limestone . The limestone increases in abundance northwestward." (Young, 1992). "The uppermost series in the Mississippian, the Chesterian Series, is present only in the southern parts of Missouri, Illinois, and Indiana. It consists of repetitive beds of sandstone, shale, and limestone that are classified into 20 formations in southern Illinois." (Young, 1992). Young, 1992 Figure 18, Hydrogeologic unit: Pennsylvanian-Mississippian-Devonian confining unit , Layer number in digital flow model: " Confining layer ".
D – Devonian rocks	Sedimentary rock aquitard (consolidated or semi-consolidated rock)	Young, 1992 Figure 4, System: "Devonian"; major rock type in Northern Missouri: " Limestone and shale ". Young, 1992 Figure 18, Northern Missouri: " Confining bed ". Young, 1992 Figure 18, Northern Missouri: " Confining bed ".
Om – Maquoketa Shale	Sedimentary rock aquitard (consolidated or semi-consolidated rock)	Young, 1992 Figure 18, Rock stratigraphic unit: " Maquoketa shale ". Northern Missouri: " Confining bed ". "This confining unit consists of the Maquoketa Shale" (Young, 1992).
Okj (Okp) – Kimmswick, Decorah, and Plattin Formation and (Joachim) Dolomite (Kimmswick formation is a minor aquifer)	Sedimentary rock aquitard (consolidated or semi-consolidated rock)	Young, 1992 Figure 18, Northern Missouri: " Decorah Fm., Plattin Fm., Joachim Dol., (confining bed) ". "The Galena Dolomite of Wisconsin is the rock stratigraphic equivalent of the Kimmswick Formation of Missouri." (Young, 1992). "In northern Missouri, the equivalent Decorah and Plattin Formations and the Joachim Dolomite (mainly carbonate rocks) contain dolomite with various amounts of interbedded shale that restricts vertical ground-water flow (Imes, 1985)." (Young, 1992).
Osp – St. Peter Sandstone	Sedimentary rock aquifer (consolidated or	Young, 1992 Figure 18, Northern Missouri: " St. Peter Ss., Everton Dol. (moderate aquifer) ". "A prominent exception to the carbonate facies is the St. Peter Sandstone—a very well

Formation name	Category	Quote
	semi-consolidated rock)	sorted, pure quartzose sandstone that is very extensive and uniform throughout the northern Midwest." (Young, 1992).
Osj – Smithville Formation; Powell, Cotter, and Jefferson City Dolomites	Sedimentary rock aquitard (consolidated or semi-consolidated rock)	"The younger Powell Dolomite and Smithville Formation in Missouri may have no equivalent to the north, presumably because of pre-St. Peter Sandstone erosion." (Young, 1992). "In northern Missouri, the Jefferson City and Cotter Dolomites , equivalent to the Willow River Member, consist primarily of fine- to medium-grained dolomite with variable amounts of chert and thin beds of shale and fine-grained sandstone ." (Young, 1992). Young, 1992 Figure 18, Northern Missouri: " Confining bed ".
Or – Roubidoux Formation	Carbonate aquifer	Young, 1992 Figure 18, Northern Missouri: "Roubidoux Fm., Gasconade Dol., Eminence Dol., Potosi Dol. (good aquifer)". "The Roubidoux Formation in Missouri, equivalent to the New Richmond Sandstone Member, consists of fine- to medium-grained, white, partly dolomitic, quartzose sandstone and fine-grained dolomite with beds of chert . The Roubidoux ranges in thickness from 100 to 250 ft and is thinnest in northeastern Missouri ." (Young, 1992).
Ogd – Gasconade Dolomite	Carbonate aquifer	Young, 1992 Figure 18, Northern Missouri: "Roubidoux Fm., Gasconade Dol. , Eminence Dol., Potosi Dol. (good aquifer)". "The Gasconade Dolomite in Missouri, equivalent to the Oneota, is coarse grained and very cherty in the lower part and fine grained and much less cherty in the upper part ." (Young, 1992).
Ee -Eminence Dolomite	Carbonate aquifer	Young, 1992 Figure 18, Northern Missouri: "Roubidoux Fm., Gasconade Dol., Eminence Dol. , Potosi Dol. (good aquifer)". "In northern Missouri-the Roubidoux Formation and the Gasconade and Eminence Dolomites -are mainly carbonate rocks that are somewhat permeable and contain some sandstone ." (Young, 1992).
Ep – Potosi Dolomite	Carbonate aquifer	Young, 1992 Figure 18, Northern Missouri: "Roubidoux Fm., Gasconade Dol., Eminence Dol., Potosi Dol. (good aquifer) ". " The most productive parts of the aquifer system in this area are the Roubidoux Formation and Gasconade, Eminence, and Potosi Dolomites (fig. 18)" (Young, 1992).
Edd – Derby-Doerun Dolomite (poor aquifer)	Carbonate aquifer	Young, 1992 Figure 18, Northern Missouri: " Derby-Doerun Dol. (poor aquifer) ". "In Missouri, Imes (1985) includes the Potosi and Derby-Doerun in his Cambrian Ordovician aquifer (fig. 18), although the Derby-Doerun is not considered to be very productive ." (Young, 1992). "In the southern and eastern parts of the area, the Potosi and Derby-Doerun Dolomites and the upper part of the Davis Formation are the equivalent rocks (mainly carbonate rocks) ." (Young, 1992).
Ed – Davis Formation	Sedimentary rock aquitard (consolidated or semi-consolidated rock)	Young, 1992 Figure 18, Northern Missouri: " The Davis Formation, Lower confining bed ". " The Davis Formation in Missouri consists of silty to sandy shale and limestone or dolomite, with beds of flat-pebble conglomerate ." (Young, 1992). "The Davis also is a dolomite but contains as much as 50 percent shale and silt ." (Young, 1992)

Formation name	Category	Quote
Eb – Bonneterre Formation	Sedimentary rock aquitard (consolidated or semi-consolidated rock)	Young, 1992 Figure 18, Northern Missouri: " Bonneterre Fm. (little information) ". " the Bonneterre Formation, form an extensive confining unit above the Mount Simon aquifer. " (Young, 1992). " The Bonneterre generally is a fine- to coarse-grained, shaly, dolomitized calcarenite with much oolitic and stromatolitic material ". "" (Young, 1992).
Eec – Eau Claire Formation	Sedimentary rock aquitard (consolidated or semi-consolidated rock)	Young, 1992 Figure 18, Northern Missouri: " Eau Claire confining unit ". " The Eau Claire Formation and its partial equivalent to the southwest, the Bonneterre Formation, form an extensive confining unit above the Mount Simon aquifer. " (Young, 1992). " The effectiveness of the Eau Claire as a confining unit depends on the relative abundance of shale, siltstone, dolomite, and sandstone in the formation. " (Young, 1992).
EI – Lamotte Sandstone	Sedimentary rock aquifer (consolidated or semi-consolidated rock)	Young, 1992 Figure 18, Northern Missouri: " Lamotte Ss. (probable aquifer) ". " The lowermost aquifer of the Cambrian-Ordovician aquifer system is composed primarily of the Mount Simon Sandstone and equivalent strata, the Lamotte Sandstone, in Missouri. " (Young, 1992). " It consists mainly of the Mount Simon Sandstone in the north and its equivalent, the Lamotte Sandstone, in northern Missouri " (Young, 1992).
pE Precambrian rocks, undivided	Endogenous bedrock	Young, 1992 Figure 4, System: " Precambrian "; major rock type in Northern Missouri: " Igneous and metamorphic crystalline rocks ". " The dense crystalline rocks of the Precambrian basement beneath the Cambrian-Ordovician aquifer system are a very effective confining unit whose upper surface marks the lower limit of the Cambrian-Ordovician aquifer system. " (Young, 1992).

Young, H.L. (1992). Hydrogeology of the Cambrian-Ordovician Aquifer System in the Northern Midwest, United States. U.S. Geological Survey Professional Paper 1405-B, 108 pp. Accessed June 14, 2022 via <https://pubs.er.usgs.gov/publication/pp1405B>

3.39 Northern Cambrian-Ordovician Aquifers, Northern Midwest Aquifer System

Supplementary Fig. 175. Hydrogeologic cross section. 20 equally spaced transparent pink bars overlie the cross section; each shaded bar depicts the vertical offset from the land surface to the top of the uppermost confining unit or endogenous bedrock.

Supplementary Fig. 176. Vertical variations in the prevalence of wells that have been defined as tapping an unconfined or a confined aquifer by the USGS. The smaller squares represent 10 m depth intervals from the land surface to 100 m; the larger squares represent 20 m intervals from 100 m to 300 m below the land surface.

The Northern Cambrian-Ordovician Aquifers subarea is located in east-central Wisconsin.

(i) A hydrogeologic cross section in Plate 1 by Young (1992) depicts sedimentary deposits overlying Precambrian bedrock.

(ii) We analysed wells within the study area that the USGS has defined as either unconfined or confined. Most (>80%) wells at depths of 160-180 m and at depths exceeding 160 m are defined as tapping a confined aquifer.

Depth to confined conditions: 160-180 m (see (ii) above)

Reference: Young, H.L. (1992). Hydrogeology of the Cambrian-Ordovician Aquifer System in the Northern Midwest, United States. US Geological Survey Professional Paper 1405-B, 108 pp. Accessed April 12, 2021 from <https://pubs.usgs.gov/pp/1405b/report.pdf> (for Plate 1 see <https://pubs.usgs.gov/pp/1405b/plate-1.pdf>)

The table below presents a series of published quotes (see quotation marks denoting text quoted from another publication, which is cited following the quotation marks with the full reference written in full below the table). The leftmost column lists a title of a hydrogeologic formation depicted in the cross section on the previous page. The rightmost column presents a quote from a hydrogeological study (see base of table for citation). The quote has been annotated with colored text to highlight how we categorized each layer (i.e., see categories in the center column in the table). Specifically: (i) blue text highlights portions of a quote that provide insights into the degree of consolidation of the formation, (ii) red text highlights portions of a quote that categorize the formation as an aquifer or an aquitard (i.e., higher versus lower permeability in the context of local hydrogeologic formations), and (iii) green text highlights portions of a quote that provide information about the lithology of the formation.

Supplementary Table 42. Hydrostratigraphy details for the Northern Cambrian Ordovician

Formation name	Category	Quote
Q – Pleistocene deposits	Unconsolidated aquifer	" unconsolidated rocks (glacial drift and alluvial deposits) and bedrock aquifers that store large quantities of ground water. " (Young, 1992). Young, 1992 Figure 4. " Unconsolidated deposits of clay, silt, sand, gravel, and boulders; degree of sorting variable"
Eem– Elk Mound Group, undifferentiated	Sedimentary rock aquitard (consolidated or semi-consolidated rock)	"Thus he proposed the term Elk Mound Group , which consists of the Mount Simon Sandstone, the Eau Claire Formation, and the Wonewoc Sandstone. " (Young, 1992). " The aquifer is generally between 50 and 150 ft thick but exceeds 200 ft on the northeastern edge of the Illinois basin. The thickness of the aquifer is uncertain north of Milwaukee, where the Ironton and Galesville Sandstones and the other sandstones of the Elk Mound Group generally cannot be differentiated in the subsurface from drill cuttings or borehole geophysical logs." (Young, 1992).
pE – Precambrian rocks, undivided	Endogenous bedrock	Young, 1992 Figure 4. System: " Precambrian "; major rock type in Northern Missouri: " Igneous and metamorphic crystalline rocks ". " The dense crystalline rocks of the Precambrian basement beneath the Cambrian-Ordovician aquifer system are a very effective confining unit whose upper surface marks the lower limit of the Cambrian-Ordovician aquifer system." (Young, 1992).

Young, H.L. (1992). Hydrogeology of the Cambrian-Ordovician Aquifer System in the Northern Midwest, United States. U.S. Geological Survey Professional Paper 1405-B, 108 pp. Accessed June 14, 2022 via <https://pubs.er.usgs.gov/publication/pp1405B>

3.40 Upper Carbonate Aquifer, Northern Midwest Aquifer System

Supplementary Fig. 177. Hydrogeologic cross section. 20 equally spaced transparent pink bars overlie the cross section; each shaded bar depicts the vertical offset from the land surface to the top of the uppermost confining unit or endogenous bedrock.

Supplementary Fig. 178. Vertical variations in the prevalence of wells that have been defined as tapping an unconfined or a confined aquifer by the USGS. The smaller squares represent 10 m depth intervals from the land surface to 100 m; the larger squares represent 20 m intervals from 100 m to 300 m below the land surface.

The Upper Carbonate Aquifer subarea is located in southeastern Minnesota and northeastern Iowa.

(i) A hydrogeologic cross section presented in Fig. 4 by Savoca et al. (1999) depicts Quaternary deposits (~50 m thick) overlying a carbonate rock aquifer (~100-150 m thick).

(ii) We analysed wells within the study area that the USGS has defined as either unconfined or confined. Most (>80%) wells at depths of 20-30 m and at depths exceeding 20 m are defined as tapping a confined aquifer.

(iii) Savoca et al. (1999) write (quote) "The Upper carbonate aquifer underlies the northern part of the study area and consists of 250 to 600 ft of Ordovician- and Devonian-age limestone, dolomite, dolomitic limestone, and shale (table 1; fig. 4, hydrogeologic section A-A') of shallow marine origin. The aquifer is overlain by unconsolidated Quaternary- and Cretaceous-age deposits (sand, gravel, and clay) and is unconfined except in areas where overlying fine-grained deposits produce locally confined conditions"

Depth to confined conditions: 20-30 m (see (ii) above)

Reference: Savoca, M.E., Sadorf, E.M., Akers, K.K. (1999). Ground-water quality in the eastern part of the Silurian-Devonian and Upper Carbonate Aquifers in the eastern Iowa basins, Iowa and Minnesota, 1996. US Geological Survey Water-Resources Investigations Report 98-4224, 35 pp. Accessed May 18, 2022 via <https://pubs.usgs.gov/wri/1998/wri984224/pdf/wri98-4224.pdf>

The table below presents a series of published quotes (see quotation marks denoting text quoted from another publication, which is cited following the quotation marks with the full reference written in full below the table). The leftmost column lists a title of a hydrogeologic formation depicted in the cross section on the previous page. The rightmost column presents a quote from a hydrogeological study (see base of table for citation). The quote has been annotated with colored text to highlight how we categorized each layer (i.e., see categories in the center column in the table). Specifically: (i) blue text highlights portions of a quote that provide insights into the degree of consolidation of the formation, (ii) red text highlights portions of a quote that categorize the formation as an aquifer or an aquitard (i.e., higher versus lower permeability in the context of local hydrogeologic formations), and (iii) green text highlights portions of a quote that provide information about the lithology of the formation.

Supplementary Table 43. Hydrostratigraphy details for the Upper Carbonate

Formation name	Category	Quote
Quaternary-age deposits	Unconsolidated aquifer	"Unconsolidated Quaternary-age deposits, which commonly contain low permeability glacial deposits, cover most of the study area (fig. 6)." (Savoca et al., 1999). "Sand, gravel, silt, and clay" (Table 1 Savoca et al., 1999). "Surficial aquifer" (Table 1, Savoca et al., 1999).
Cretaceous-age rocks, undifferentiated	Clastic sedimentary rock aquifer (consolidated or semi-consolidated rock)	"Cretaceous-age deposits (sand, gravel, and clay)" (Savoca et al., 1999). "Dakota Formation" (Table 1 Savoca et al., 1999). "Isolated, water-bearing units" (Table 1, Savoca et al., 1999).
Upper Carbonate and Silurian-Devonian aquifer	Carbonate aquifer	"Limestone and dolomite are the dominant lithologies in the Silurian-Devonian and Upper Carbonate aquifers." (Savoca et al., 1999). "The Upper Carbonate aquifer underlies the northern part of the study area and consists of 250 to 600 ft of Ordovician- and Devonian-age limestone, dolomite, dolomitic limestone, and shale (table 1; fig. 4, hydrogeologic section A-A') of shallow marine origin." Savoca et al., 1999).
Decorah Shale, Platteville Formation, and Glenwood Shale, Maquoketa Shale	Sedimentary rock aquitard (consolidated or semi-consolidated rock)	"The Upper Carbonate aquifer is underlain by a confining unit consisting of the Ordovician-age Decorah Shale, Platteville Formation, and Glenwood Shale." (Savoca et al., 1999). "Shale, dolomitic limestone, and limestone" & "Confining unit" (Table 1, Savoca et al., 1999).
Ordovician-age rock, undifferentiated	Sedimentary rock aquitard (consolidated or semi-consolidated rock)	"The Silurian-Devonian aquifer is underlain by a confining unit consisting of the Ordovician-age Maquoketa Shale." (Savoca et al., 1999). "Shale, dolomitic limestone, and limestone" & "Confining unit" (Table 1, Savoca et al., 1999).

Savoca, M.E., Sadorf, E.M., Akers, K.K. (1999). Ground-Water quality in the eastern part of the Silurian-Devonian and Upper Carbonate Aquifers in the eastern Iowa basins, Iowa and Minnesota, 1996. US Geological Survey Water-Resources Investigations Report 98-4224, 35 pp. Accessed June 5, 2022 via <https://pubs.er.usgs.gov/publication/wri984224>

3.41 Western Cambrian-Ordovician Aquifers, Northern Midwest Aquifer System

Supplementary Fig. 179. Hydrogeologic cross section. 20 equally spaced transparent pink bars overlie the cross section; each shaded bar depicts the vertical offset from the land surface to the top of the uppermost confining unit or endogenous bedrock.

Supplementary Fig. 180. Vertical variations in the prevalence of wells that have been defined as tapping an unconfined or a confined aquifer by the USGS. The smaller squares represent 10 m depth intervals from the land surface to 100 m; the larger squares represent 20 m intervals from 100 m to 300 m below the land surface.

The Western Cambrian-Ordovician Aquifers area is situated in southeastern Minnesota and western Wisconsin.

(i) A hydrogeologic cross section presented in Fig. 5 by Seaberg (2000) depicts layered sedimentary sequences including discontinuous aquitards (Glenwood-Platteville-Decorah units) and more continuous aquitards (e.g., St. Lawrence Confining Bed).

(ii) We analysed wells within the study area that the USGS has defined as either unconfined or confined. Most (>80%) wells at depths of 30-40 m and at depths exceeding 30 m are defined as tapping a confined aquifer.

Depth to confined conditions: 30-40 m (see (ii) above)

Reference: Seaberg, J.K. (2000). Overview of the Twin Cities Metropolitan Groundwater Model. Minnesota Pollution Control Agency Report, 65 pp. Accessed May 18, 2022 via <https://www.pca.state.mn.us/sites/default/files/mm-overview.pdf>

The table below presents a series of published quotes (see quotation marks denoting text quoted from another publication, which is cited following the quotation marks with the full reference written in full below the table). The leftmost column lists a title of a hydrogeologic formation depicted in the cross section on the previous page. The rightmost column presents a quote from a hydrogeological study (see base of table for citation). The quote has been annotated with colored text to highlight how we categorized each layer (i.e., see categories in the center column in the table). Specifically: (i) blue text highlights portions of a quote that provide insights into the degree of consolidation of the formation, (ii) red text highlights portions of a quote that categorize the formation as an aquifer or an aquitard (i.e., higher versus lower permeability in the context of local hydrogeologic formations), and (iii) green text highlights portions of a quote that provide information about the lithology of the formation.

Supplementary Table 44. Hydrostratigraphy details for the Western Cambrian-Ordovician

Formation name	Category	Quote
Drift	Unconsolidated aquifer	Aquifer layer Young, H.L. (1992) "Because glacial drift is almost universally present in the study area and is the shallowest permeable rock material, it provides a ready source of ground water. Ground-water availability from the drift is directly proportional to the amount of permeable, well-sorted sand and gravel within the drift . The large variety of sediment types in the drift have a wide range of sorting and depositional form" Young, H.L. (1992). " Sand and gravel is less abundant, present as discontinuous lenses within or beneath moraine or as more extensive outwash and ice-contact deposits. " Young, H.L. (1992)
Glenwood-Platteville-Decorah confining bed	Sedimentary rock aquitard (consolidated or semi-consolidated rock)	Confining layer Young, H.L. (1992) "This confining unit consists of the Maquoketa Shale, the Galena Dolomite, and the Decorah, Platteville, and Glenwood Formations or equivalents. " Young, H.L. (1992).
St. Peter aquifer	Sedimentary rock aquifer (consolidated or semi-consolidated rock)	"Underlying the Glenwood Shale is the St. Peter Sandstone, a fine to medium-grained, well-sorted, white to buff quartz sandstone occupying approximately 1,760 square kilometers (680 square miles) of the seven-county metropolitan area, as determined from Mossler and Tipping (2000)." (Seaberg, 2000). Aquifer layer Young, H.L. (1992)
Prairie du Chien-Jordan aquifer	Sedimentary rock aquifer (consolidated or semi-consolidated rock)	Aquifer layer Young, H.L. (1992) "Although this multiunit aquifer may be the least uniform of the bedrock aquifer units in the northern Midwest (fig. 18), it is a major source of ground water in Iowa and Minnesota. In these States, the Jordan Sandstone and all or parts of the overlying Prairie du Chien Group are in direct hydraulic connection, resulting in a highly productive aquifer." Young, H.L. (1992).
St. Lawrence confining bed	Sedimentary rock aquitard (consolidated or semi-consolidated rock)	Confining layer Young, H.L. (1992) "The St. Lawrence and Franconia Formations form an important regional confining unit over the Ironton-Galesville aquifer. Although these formations are dominantly sandstone in the northern part of the area, they are very silty and shaly, fine grained, poorly sorted, and dolomitic. Thus, the units are anisotropic and restrict vertical movement of ground water. " Young, H.L. (1992).
Franconia-Ironton-Galesville aquifer	Sedimentary rock aquifer (consolidated or	Aquifer layer Young, H.L. (1992) "The Ironton and Galesville Sandstones form the most important aquifer of the Cambrian-Ordovician aquifer system in the east-central part of the study area, although they generally are not the only rock

Formation name	Category	Quote
	semi-consolidated rock	units open to deep wells." Young, H.L. (1992). "The aquifer terminates to the west, south, and east (fig. 10) as the sandstones grade into carbonate rocks, primarily dolomite. " Young, H.L. (1992). Franconian: "not generally regarded as an aquifer by itself" (Seaberg, 2000).
Eau Claire confining bed	Sedimentary rock aquitard (consolidated or semi-consolidated rock)	" Confining layer " Young, H.L. (1992) "The Eau Claire Formation and its partial equivalent to the southwest, the Bonneterre Formation, form an extensive confining unit above the Mount Simon aquifer. The effectiveness of the Eau Claire as a confining unit depends on the relative abundance of shale, siltstone, dolomite, and sandstone in the formation. " Young, H.L. (1992).
Mt. Simon-Hinckley aquifer	Sedimentary rock aquifer (consolidated or semi-consolidated rock)	" Aquifer layer " Young, H.L. (1992) "The lowermost aquifer of the Cambrian-Ordovician aquifer system is composed primarily of the Mount Simon Sandstone and equivalent strata , the Lamotte Sandstone, in Missouri. In Minnesota, the Mount Simon is underlain by Precambrian sedimentary rocks—the Hinckley Sandstone and the older Fond du Lac Formation." Young, H.L. (1992).
Undifferentiated Precambrian bedrock	Endogenous bedrock	" Impermeable boundary " Young, H.L. (1992) "The lowermost hydrogeologic unit is a confining unit that consists of generally very low permeability crystalline rocks of the Precambrian basement and, in Minnesota, most of the Precambrian sedimentary rocks below the Hinckley Sandstone. " Young, H.L. (1992).

Young, H.L. (1992). Hydrogeology of the Cambrian-Ordovician Aquifer System in the Northern Midwest, United States. U.S. Geological Survey Professional Paper 1405-B, 108 pp. Accessed June 17, 2022 via <https://pubs.usgs.gov/pp/1405b/report.pdf>

Seaberg, J.K. (2000). Overview of the Twin Cities Metropolitan Groundwater Model. Minnesota Pollution Control Agency Report, 65 pp. Accessed May 18, 2022 via <https://www.pca.state.mn.us/sites/default/files/mm-overview.pdf>

3.42 Mesilla Valley, Rincon-Mesilla Valleys

Supplementary Fig. 181. Hydrogeologic cross section. 20 equally spaced transparent pink bars overlie the cross section; each shaded bar depicts the vertical offset from the land surface to the top of the uppermost confining unit or endogenous bedrock.

Supplementary Fig. 182. Vertical variations in the prevalence of wells that have been defined as tapping a confined or a confined aquifer by the USGS. The smaller squares represent 10 m depth intervals from the land surface to 100 m; the larger squares represent 20 m intervals from 100 m to 300 m below the land surface.

The Mesilla Valley is located in southern New Mexico.

(i) A hydrogeologic cross section presented in Fig. 3 by Robertson et al. (2022) does not depict a clear confining unit within the shallow portion of the aquifer system.

(ii) We analysed wells within the study area that the USGS has defined as either unconfined or confined. Most (>80%) wells at depths of 240-260 m and at depths exceeding 240 m are defined as tapping a confined aquifer.

Depth to confined conditions: 240-260 m (see (ii) above)

Reference: Robertson, A.J., Matherne, A.M., Pepin, J.D., Ritchie, A.B., Sweetkind, D.S., Teeple, A.P., Granados-Olivas, A., García-Vásquez, A.C., Carroll, K.C., Fuchs, E.H., Galanter, A.E. (2022). Mesilla/Conejos-Médanos Basin: US-Mexico Transboundary Water Resources. *Water*, 14, 134.

The table below presents a series of published quotes (see quotation marks denoting text quoted from another publication, which is cited following the quotation marks with the full reference written in full below the table). The leftmost column lists a title of a hydrogeologic formation depicted in the cross section on the previous page. The rightmost column presents a quote from a hydrogeological study (see base of table for citation). The quote has been annotated with colored text to highlight how we categorized each layer (i.e., see categories in the center column in the table). Specifically: (i) blue text highlights portions of a quote that provide insights into the degree of consolidation of the formation, (ii) red text highlights portions of a quote that categorize the formation as an aquifer or an aquitard (i.e., higher versus lower permeability in the context of local hydrogeologic formations), and (iii) green text highlights portions of a quote that provide information about the lithology of the formation.

Supplementary Table 45. Hydrostratigraphy details for the Mesilla Valley

Formation name	Category	Quote
Rio Grande alluvium	Unconsolidated aquifer	"A thin layer (< 25 m) of unconsolidated Quaternary alluvial and fluvial deposits , known as the Rio Grande alluvium, overlies the Santa Fe" Robertson et al., 2022). "The Rio Grande alluvium is composed of river-channel and overbank depositional facies ranging in texture from sand and gravel to silt and clay that are generally 15 to 38 m thick , respectively [11]." Robertson et al., 2022). " Groundwater levels in the Rio Grande alluvium are shallow and unconfined and generally decrease from north to south at an average gradient of about 0.8 to 1.1 m/km [38], closely following the topographic gradient. " (Robertson et al., 2022).
Upper Santa Fe	Unconsolidated aquifer	"The upper Santa Fe unit is about 3 to 4 Ma and consists of fluvial deposits from a large, braided river of the ancestral Rio Grande, with channel sands and gravels from as far north as the mountains in southern Colorado and alluvial fan deposits derived from basin-bounding highlands (Figure 1) [8]." Robertson et al., 2022). "Hawley and Kennedy (2004) estimated the "most productive" portion of the aquifer system (the Rio Grande alluvium, upper Santa Fe unit , and middle Santa Fe unit) to hold about 17,300 hm ³ of available freshwater (<1000 mg/L DS) [8]." Robertson et al., 2022).
Middle Santa Fe	Unconsolidated aquifer	"The middle Santa Fe unit is composed of aeolian dune sequences up to 610 m thick that intertongue with alluvial deposits near the bounding mountains and fluvial and playa-lake deposits in the inner Basin [8]." (Robertson et al., 2022). "Hawley and Kennedy (2004) estimated the "most productive" portion of the aquifer system (the Rio Grande alluvium, upper Santa Fe unit, and middle Santa Fe unit) to hold about 17,300 hm ³ of available freshwater (<1000 mg/L DS) [8]." (Robertson et al., 2022).
Lower Santa Fe	Clastic sedimentary rock aquifer (consolidated or semi-consolidated rock)	"The lower Santa Fe unit is primarily fine-grained and partly consolidated with some calcium-sulfate and sodium-sulfate evaporites and cementation, which was deposited in a closed basin." (Robertson et al., 2022). (Robertson et al., 2022). "specific capacity estimates of the middle Santa Fe are usually less than 8 L/s/m and between 0.2 to 2 L/s/m for the lower Santa Fe [8]."

Formation name	Category	Quote
Basement	Endogenous bedrock	"The pre-Santa Fe Group rocks are deformed and faulted" (Hanson et al., 2020). "The basement units in the TRG were grouped into seven groups that represent the bedrock units, Tertiary sediments, intrusive rocks, and volcanics (table 4; Sweetkind, 2017, fig. 23)" (Hanson et al., 2020).

Robertson, A.J., Matherne, A.M., Pepin, J.D., Ritchie, A.B., Sweetkind, D.S., Teeple, A.P., Granados-Olivas, A., García-Vásquez, A.C., Carroll, K.C., Fuchs, E.H., Galanter, A.E. (2022). Mesilla/Conejos-Médanos Basin: US-Mexico Transboundary Water Resources. *Water*, 14, 134.

Hanson, R.T.; Ritchie, A.B.; Boyce, S.E.; Galanter, A.E.; Ferguson, I.A.; Flint, L.E., Henson, W.R. (2020). Rio Grande transboundary integrated hydrologic model and water-availability analysis, New Mexico and Texas, United States, and Northern Chihuahua, Mexico. In *US Geological Survey Scientific Investigations Report 2019-5120*; U.S. Geological Survey. Accessed June 5, 2022 via <https://pubs.er.usgs.gov/publication/sir20195120>

3.43 Lower Santa Ynez Valley, Santa Ynez Valley

Supplementary Fig. 183. Hydrogeologic cross section. 20 equally spaced transparent pink bars overlie the cross section; each shaded bar depicts the vertical offset from the land surface to the top of the uppermost confining unit or endogenous bedrock.

Supplementary Fig. 184. Vertical variations in the prevalence of wells that have been defined as tapping an unconfined or a confined aquifer by the USGS. The smaller squares represent 10 m depth intervals from the land surface to 100 m; the larger squares represent 20 m intervals from 100 m to 300 m below the land surface.

The Lower Santa Ynez Valley is located in western California.

(i) A hydrogeologic cross section presented in Fig. 2a.2-5a by the Western Management Area Groundwater Sustainability Agency (2022) suggests that the uppermost confining unit is

typically 202 meters below land surface (25th-75th percentile range: 56-243 meters below land surface).

(ii) We analysed wells within the study area that the USGS has defined as either unconfined or confined. All n=4 wells (with depths ranging from 49 m to 117 m) are classified as unconfined.

(iii) Regarding confined conditions, the Western Management Area Groundwater Sustainability Agency (2022) states (quote) “The main zone throughout most of the Lompoc Plain subarea is separated from the middle zone by lenses of silt and clay that result in confined or partially confined conditions in the main zone. However, in the eastern, southern, and northern portions of the Lompoc Plain subarea, the confining deposits are less continuous or absent...”

Depth to confined conditions: 200-220 m based on (i) and (iii)

References: Western Management Area Groundwater Sustainability Agency (2022). Groundwater sustainability plan for the Santa Ynez River Valley groundwater basin, Bulletin 118 Basin No. 3-15, Western Management Area Groundwater Sustainability Agency Report, 1413 pp. Accessed April 11, 2022 via https://www.santaynezwater.org/files/273c6a5e1/SYRVGB+SG+MA+GSP+-+WMA+JAN+2022_compressed.pdf

The table below presents a series of published quotes (see quotation marks denoting text quoted from another publication, which is cited following the quotation marks with the full reference written in full below the table). The leftmost column lists a title of a hydrogeologic formation depicted in the cross section on the previous page. The rightmost column presents a quote from a hydrogeological study (see base of table for citation). The quote has been annotated with colored text to highlight how we categorized each layer (i.e., see categories in the center column in the table). Specifically: (i) blue text highlights portions of a quote that provide insights into the degree of consolidation of the formation, (ii) red text highlights portions of a quote that categorize the formation as an aquifer or an aquitard (i.e., higher versus lower permeability in the context of local hydrogeologic formations), and (iii) green text highlights portions of a quote that provide information about the lithology of the formation.

Supplementary Table 46. Hydrostratigraphy details for the Lower Santa Ynez Valley

Formation name	Category	Quote
Upper Aquifer- Younger Alluvium	Unconsolidated aquifer	"Groundwater occurs in thin, unconsolidated sedimentary layers of younger alluvium directly over non-water-bearing, consolidated geologic units (Section 2a.2)." (WMAGSA, 2020). "an Upper Aquifer, consisting of younger alluvial sediments that are primarily associated with river and surface water geomorphic processes." (WMAGSA, 2020). "a Lower Aquifer, which is more extensive throughout the Basin and consists of older geologic depositions." (WMAGSA, 2020).
Lower Aquifer – Careaga Sand Tca	Unconsolidated aquifer (unconfined aquifer, and semi- confined aquifer)	"For groundwater management purposes, two principal aquifers were defined based on the Lompoc Plain location: the Upper Aquifer, which consists of alluvial sediments, and the Lower Aquifer, which consists of the water-bearing Careaga Sand and Paso Robles Formation." (WMAGSA, 2020). "The Santa Rita Upland contains unconsolidated water-bearing principal aquifer units of the Lower Aquifer within an east/west-trending geologic syncline fold." (WMAGSA, 2020).
Consolidated Rock-non-water bearing	Clastic sedimentary rock aquifer (consolidated or semi-consolidated rock)	"Non-water-bearing consolidated geologic units also form the lateral boundaries as exposed bedrock in this area." (WMAGSA, 2020). "Tertiary-Mesozoic Rocks are consolidated non-water bearing units, all of marine origin. They consist of the near-shore marine Foxen (Tf), Sisquoc (Tsq), and Monterey (Tm) Formations." (WMAGSA, 2020).

Western Management Area Groundwater Sustainability Agency (2020). Groundwater sustainability plan for the Santa Ynez River Valley groundwater basin bulletin 118 basin no. 3-15 western management area groundwater sustainability agency.
https://www.santaynezwater.org/files/273c6a5e1/SYRVGB+SGMA+GSP+-+WMA+JAN+2022_compressed.pdf

3.44 Boise Valley and Homedale Murphy Area, Western Snake River Plain

Supplementary Fig. 185. Hydrogeologic cross section. 20 equally spaced transparent pink bars overlie the cross section; each shaded bar depicts the vertical offset from the land surface to the top of the uppermost confining unit or endogenous bedrock.

Supplementary Fig. 186. Vertical variations in the prevalence of wells that have been defined as tapping an unconfined or a confined aquifer by the USGS. The smaller squares represent 10 m depth intervals from the land surface to 100 m; the larger squares represent 20 m intervals from 100 m to 300 m below the land surface.

The Boise Valley and Homedale Murphy Area is located in Idaho.

(i) A hydrogeologic cross section presented in Fig. 39 by Lindholm (1996) suggests that the upper unit (primarily sand and gravel) is underlain by a confining unit ("Middle unit").

(ii) We analysed wells within the study area that the USGS has defined as either unconfined or confined. Two of the three deepest wells in the dataset (depths range from 209 m to 372 m) as classified as unconfined; the deepest well (372 m deep) is classified as confined. The available USGS well data are insufficient to evaluate the depths at which the aquifer system transitions from unconfined to confined conditions.

Depth to confined conditions:
>260 m (see (ii) above)

Reference: Lindholm, G. F. (1996). Summary of the Snake River Plain regional aquifer-system analysis in Idaho and eastern Oregon. U.S Geological Survey Professional Paper 1408-A. 59 pp. Accessed March 1, 2021 via <https://pubs.usgs.gov/pp/1408a/report.pdf>

The table below presents a series of published quotes (see quotation marks denoting text quoted from another publication, which is cited following the quotation marks with the full reference written in full below the table). The leftmost column lists a title of a hydrogeologic formation depicted in the cross section on the previous page. The rightmost column presents a quote from a hydrogeological study (see base of table for citation). The quote has been annotated with colored text to highlight how we categorized each layer (i.e., see categories in the center column in the table). Specifically: (i) blue text highlights portions of a quote that provide insights into the degree of consolidation of the formation, (ii) red text highlights portions of a quote that categorize the formation as an aquifer or an aquitard (i.e., higher versus lower permeability in the context of local hydrogeologic formations), and (iii) green text highlights portions of a quote that provide information about the lithology of the formation.

Supplementary Table 47. Hydrostratigraphy details for Boise Valley and Homedale Murphy Area

Formation name	Category	Quote
Upper unit: Sand and gravel	Unconsolidated aquifer	"The most productive aquifers in the 4,800-square-mile western plain are alluvial sand and gravel in the Boise River valley." (Lindholm, 1996). " Alluvium is thickest (several hundred feet) and the percentage of sand and gravel largest in the Boise River valley." (Lindholm, 1996).
Middle unit: Sedimentary Unit	Clastic sedimentary rock aquifer (consolidated or semi-consolidated rock)	"Pre-Tertiary sedimentary rocks in the eastern part of the basin (fig. 8) are generally a complex of shale, argillite, sandstone, and limestone ." (Lindholm, 1996). "Temporal changes in ground-water solute concentrations in the Snake River Plain are poorly defined. Low (1987) noted that concentrations of dissolved solids and chloride are generally greatest in areas of fine-grained sedimentary rocks and in intensively irrigated areas." (Lindholm, 1996).
Lower Unit: Volcanic rock	Volcanic aquifer	" Tertiary volcanic rocks east of the batholith range from rhyodacite to basalt ." (Lindholm, 1996). " Very low; confining unit High; may be extremely high Horizontal: low to moderate, depends on vesicularity and degree of fracturing Vertical: depends on degree of fracturing; commonly several orders of magnitude lower than horizontal conductivity " (Lindholm, 1996).

Lindholm, G. F. (1996). Summary of the Snake River Plain regional aquifer-system analysis in Idaho and eastern Oregon. U.S Geological Survey Professional Paper 1408-A. 59 pp. Accessed March 1, 2021 via <https://pubs.usgs.gov/pp/1408a/report.pdf>

3.45 Mountain Home Plateau, Western Snake River Plain

Supplementary Fig. 187. Hydrogeologic cross section. 20 equally spaced transparent pink bars overlie the cross section; each shaded bar depicts the vertical offset from the land surface to the top of the uppermost confining unit or endogenous bedrock.

Supplementary Fig. 188. Vertical variations in the prevalence of wells that have been defined as tapping an unconfined or a confined aquifer by the USGS. The smaller squares represent 10 m depth intervals from the land surface to 100 m; the larger squares represent 20 m intervals from 100 m to 300 m below the land surface.

The Mountain Home Plateau subarea of the broader Western Snake River Plain is located in southwestern Idaho.

(i) A hydrogeologic cross section presented in Fig. 4 by Norton et al. (1982) for the Mountain Home Plateau does not depict a clear confining unit. The median depth to a confined unit exceeds 336 meters below land surface (median of pink transparent bars in cross section; the 25th-75th percentile range is >336 m to >351 m).

(ii) There is only n=1 well (with a depth of 112 m) that the USGS has defined as tapping either a confined or unconfined aquifer; this single data point is insufficient to evaluate the depths at which the aquifer system transitions from unconfined to confined conditions.

Depth to confined conditions: >336 m (based on (i))

Reference: Norton, M.A., Ondrechen, W., Baggs, J.L., (1982). Ground water investigation of the Mountain Home Plateau, Idaho. Idaho Department of Water Resources report AN/KW C-4380 200, 67 pp. Accessed May 19, 2022 via <https://idwr.idaho.gov/wp-content/uploads/sites/2/publications/198208-OFR-gw-investigations-mthome-plateau-id.pdf>

The table below presents a series of published quotes (see quotation marks denoting text quoted from another publication, which is cited following the quotation marks with the full reference written in full below the table). The leftmost column lists a title of a hydrogeologic formation depicted in the cross section on the previous page. The rightmost column presents a quote from a hydrogeological study (see base of table for citation). The quote has been annotated with colored text to highlight how we categorized each layer (i.e., see categories in the center column in the table). Specifically: (i) blue text highlights portions of a quote that provide insights into the degree of consolidation of the formation, (ii) red text highlights portions of a quote that categorize the formation as an aquifer or an aquitard (i.e., higher versus lower permeability in the context of local hydrogeologic formations), and (iii) green text highlights portions of a quote that provide information about the lithology of the formation.

Supplementary Table 48. Hydrostratigraphy details for the Mountain Home Plateau

Formation name	Category	Quote
Sediment	Unconsolidated aquifer	" alluvium and younger terrace gravels " (Norton et al., 1982). " Unconsolidated clay, silt, sand, and gravel occurring beneath flood plains of Boise and Snake Rivers." (Norton et al., 1982). " Hydraulic conductivity generally high : however, because of thinness and irregularity of beds, yields to wells are generally small to moderate." (Norton et al., 1982).
Basalt (Bruneau Formation of Idaho Group)	Volcanic aquifer	"The basalts of the Bruneau Formation thin rapidly to the east (cross section A)" (Norton et al., 1982). "Includes fan deposits consisting largely of coarse sands derived from decayed granitic rocks . Thickness of fan deposit does not exceed 300 ft. Also includes vesicular olivine basalt, dark gray to black, weathers to reddish-gray-brown ." (Norton et al., 1982). "Fan deposits are generally above water table. Basalt composes principal aquifer in Mountain Home area . Reported well-yields from basalt range from 10 to 3,500 gal/min. Detrital material generally has low hydraulic conductivity ." (Norton et al., 1982).
Sediment (Glenns Ferry Formation of Idaho Group)	Sedimentary rock aquitard (consolidated or semi-consolidated rock)	" Poorly consolidated detrital material and minor flows of olivine basalt. Includes lake and stream deposits consisting of massive silt layers, cemented sand beds, thin beds of dark clay, olive silt, and granitic sand and fine pebble gravel ." (Norton et al., 1982). " Hydraulic conductivity generally low ." (Norton et al., 1982).

Norton, M.A., Ondrechen, W., Baggs, J.L. (1982). Ground water investigation of the Mountain Home Plateau, Idaho. Idaho Department of Water Resources report AN/KW C-4380 200. 67 pp. Accessed May 19, 2022 via <https://idwr.idaho.gov/wp-content/uploads/sites/2/publications/198208-OFR-gw-investigations-mthome-plateau-id.pdf>

3.46 Antelope Valley

Supplementary Fig. 189. Hydrogeologic cross section. 20 equally spaced transparent pink bars overlie the cross section; each shaded bar depicts the vertical offset from the land surface to the top of the uppermost confining unit or endogenous bedrock.

Supplementary Fig. 190. Vertical variations in the prevalence of wells that have been defined as tapping a confined or a confined aquifer by the USGS. The smaller squares represent 10 m depth intervals from the land surface to 100 m; the larger squares represent 20 m intervals from 100 m to 300 m below the land surface.

The Antelope Valley is located in southcentral California.

(i) A hydrogeologic cross section presented in Fig. 3 by Duell Jr. (1987) suggests that shallow unconsolidated deposits are underlain by a confining unit comprised of lacustrine clay.

(ii) We analysed wells within the study area that the USGS has defined as either unconfined or confined. Most (>80%) wells at depths of x m and at depths exceeding y m are defined as tapping a confined aquifer.

Depth to confined conditions: 120-140 m (see (ii) above)

Reference: Duell Jr., L.F.W. (1987). Geohydrology of the Antelope Valley area, California, and design for a ground-water-quality monitoring network. US Geological Survey Water-Resources Investigations Report 84-4081, 76 pp. Accessed April 18, 2022 via <https://pubs.usgs.gov/wri/1984/4081/report.pdf>

The table below presents a series of published quotes (see quotation marks denoting text quoted from another publication, which is cited following the quotation marks with the full reference written in full below the table). The leftmost column lists a title of a hydrogeologic formation depicted in the cross section on the previous page. The rightmost column presents a quote from a hydrogeological study (see base of table for citation). The quote has been annotated with colored text to highlight how we categorized each layer (i.e., see categories in the center column in the table). Specifically: (i) blue text highlights portions of a quote that provide insights into the degree of consolidation of the formation, (ii) red text highlights portions of a quote that categorize the formation as an aquifer or an aquitard (i.e., higher versus lower permeability in the context of local hydrogeologic formations), and (iii) green text highlights portions of a quote that provide information about the lithology of the formation.

Supplementary Table 49. Hydrostratigraphy details for the Antelope Valley

Formation name	Category	Quote
Unconsolidated deposits	Unconsolidated aquifer	" The unconsolidated deposits that underlie Antelope Valley (pi. 1) include younger and older alluvium, older fan deposits, windblown dune sand, and playa deposits. These deposits comprise the aquifers of the area. " Duell, (1987). "The older alluvium is porous and permeable and yields water freely, and is the most important water-bearing unit. " Duell, (1987).
Lacustrine deposits	Sedimentary rock aquitard (consolidated or semi-consolidated rock)	" Playa or lacustrine deposits of Pliocene through Holocene age are composed of siltstone, clay, and marl . During pluvial periods, or times of relatively heavy precipitation, massive beds of blue clay formed in deep, perennial lakes. Individual clay beds are locally as much as 400 feet thick. These beds are interbedded with lenses of coarser material as much as 20 feet thick. The clay yields virtually no water to wells, but interbedded materials supply some water to wells. " Duell, (1987).
Consolidated rocks	Sedimentary rock aquitard (consolidated or semi-consolidated rock)	" Consolidated rocks surround Antelope Valley and form the sides and bottom of the ground-water basin (pi. 1)." Duell, (1987). " Consolidated sedimentary rocks of Tertiary age yield little water, if any. " Duell, (1987).

Duell, L. F. (1987). Geohydrology of the Antelope Valley Area, California and design for a ground-water-quality monitoring network (Vol. 84, No. 4081). Department of the Interior, US Geological Survey. <https://pubs.er.usgs.gov/publication/wri844081>

3.47 Big Bear Valley

Supplementary Fig. 191. Hydrogeologic cross section. 20 equally spaced transparent pink bars overlie the cross section; each shaded bar depicts the vertical offset from the land surface to the top of the uppermost confining unit or endogenous bedrock.

Big Bear Valley is located in southcentral California.

(i) A hydrogeologic cross section presented in Fig. 6 by Flint et al. (2012) suggests that the aquifer system does not host a clear confining unit within the aquifer system. Pre-Tertiary basement rocks underlie the Quaternary and Tertiary deposits; the typical (i.e., median) depth to these basement rocks as depicted in the cross section is 282 meters below the land surface (see pink shaded bars in the cross section to the left).

(ii) We did not identify sufficient wells within the valley for which the USGS has defined the well as tapping an unconfined or confined aquifer.

Depth to confined conditions:
280-300 m (see (i) above)

Reference: Flint, L.E., Martin, P., eds. (2012). Geohydrology of Big Bear Valley, California: Phase 1—Geologic Framework, Recharge, and Preliminary Assessment of the Source and Age of Groundwater; U.S. Geological Survey Scientific Investigations Report 2012–5100, 112 pp. Accessed November 28, 2021 from <https://pubs.usgs.gov/sir/2012/5100/pdf/sir20125100.pdf>

The table below presents a series of published quotes (see quotation marks denoting text quoted from another publication, which is cited following the quotation marks with the full reference written in full below the table). The leftmost column lists a title of a hydrogeologic formation depicted in the cross section on the previous page. The rightmost column presents a quote from a hydrogeological study (see base of table for citation). The quote has been annotated with colored text to highlight how we categorized each layer (i.e., see categories in the center column in the table). Specifically: (i) blue text highlights portions of a quote that provide insights into the degree of consolidation of the formation, (ii) red text highlights portions of a quote that categorize the formation as an aquifer or an aquitard (i.e., higher versus lower permeability in the context of local hydrogeologic formations), and (iii) green text highlights portions of a quote that provide information about the lithology of the formation.

Supplementary Table 50. Hydrostratigraphy details for the Big Bear Valley

Formation name	Category	Quote
Quaternary and Tertiary deposits	Unconsolidated aquifer	" Most of the water supply for the Big Bear area is pumped from the unconsolidated Quaternary alluvial deposits in the groundwater basin. The water-bearing deposits in the groundwater basin have been classified into upper, middle, and lower aquifers (GeoScience Support Services, Inc., 1999), and the upper and middle aquifers are the primary water producers. Inspection of geologic logs from wells drilled in the Baldwin Lake area indicates that the deposits consist of sands and gravels with interbedded clays. " (Flint et al., 2012)
Pre-Tertiary basement rocks	Endogenous bedrock	" The basement rocks are dominated by (1) large Cretaceous granitic bodies ranging in composition from monzogranite to gabbro, (2) metamorphosed sedimentary rocks ranging in age from late Paleozoic to late Proterozoic, and (3) Middle Proterozoic gneiss (Miller, 2004). These rocks are complexly deformed by normal, reverse, and thrust faults, and are tightly folded in some places (Miller, 2004). In general, the basement rocks are of low permeability and are not considered a major water-bearing unit except in fractures and weathered zones that can create shallow perched groundwater zones. " (Flint et al., 2012)

Flint, L.E., Martin, P., eds., with contributions by Brandt, J., Christensen, A.H., Flint, A.L., Flint, L.E., Hevesi, J.A., Jachens, R., Kulongoski, J.T., Martin, P., Sneed, M. (2012). Geohydrology of Big Bear Valley, California: Phase 1—Geologic Framework, Recharge, and Preliminary Assessment of the Source and Age of Groundwater: U.S. Geological Survey Scientific Investigations Report 2012–5100, 112 p.

3.48 Bighorn Basin

Supplementary Fig. 192. Hydrogeologic cross section. 20 equally spaced transparent pink bars overlie the cross section; each shaded bar depicts the vertical offset from the land surface to the top of the uppermost confining unit or endogenous bedrock.

Supplementary Fig. 193. Vertical variations in the prevalence of wells that have been defined as tapping an unconfined or a confined aquifer by the USGS. The smaller squares represent 10 m depth intervals from the land surface to 100 m; the larger squares represent 20 m intervals from 100 m to 300 m below the land surface.

The Bighorn Basin is located in central Wyoming.

(i) A hydrogeologic cross section presented in Plate VI by Tauchen et al. (2012) suggests that the aquifer system does not have a single and continuous confining unit at shallow depths.

(ii) We analysed wells within the study area that the USGS has defined as either unconfined or confined. Most (>80%) wells at depths of 70-80 m and at depths exceeding 120 m are defined as tapping a confined aquifer.

Depth to confined conditions: 70-80 m (see (ii) above)

Reference: Tauchen, P., Bartos, T.T., Clarey, K.E., Quillinan, S.A., Hallberg, L.L., Clark, M.L., Thompson, M., Gribb, N., Worman, B., Gracias, T. (2012). Wind/Bighorn River Basin Water Plan Update Groundwater Study Level 1 (2008–2011). Groundwater Determination. Wyoming Water Development Commission Technical Memorandum, 397 pp. Accessed March 29, 2021 from https://waterplan.state.wy.us/pla n/bighorn/2010/gw-finalrept/gw_toc.html

The table below presents a series of published quotes (see quotation marks denoting text quoted from another publication, which is cited following the quotation marks with the full reference written in full below the table). The leftmost column lists a title of a hydrogeologic formation depicted in the cross section on the previous page. The rightmost column presents a quote from a hydrogeological study (see base of table for citation). The quote has been annotated with colored text to highlight how we categorized each layer (i.e., see categories in the center column in the table). Specifically: (i) blue text highlights portions of a quote that provide insights into the degree of consolidation of the formation, (ii) red text highlights portions of a quote that categorize the formation as an aquifer or an aquitard (i.e., higher versus lower permeability in the context of local hydrogeologic formations), and (iii) green text highlights portions of a quote that provide information about the lithology of the formation.

Supplementary Table 51. Hydrostratigraphy details for the Bighorn Basin

Formation name	Category	Quote
Tu (Tertiary rocks undivided (Post-Fort Union Formation) (confining unit exists))	Clastic sedimentary rock aquifer (consolidated or semi-consolidated rock)	"Below the major alluvial aquifer at the land surface and above the thick, widespread Upper Cretaceous Cody [Major] confining unit are three Tertiary or Upper Cretaceous major (mostly sandstone) aquifers (Fort Union, Lance, Mesaverde) and one major confining unit (Meeteetse), all components of the lower Tertiary/Upper Cretaceous aquifer system ; the aquifers are generally accessible in the central basins." Tauchen et al. (2012) "Aquifers within the system are lenticular, discontinuous sandstone bodies that are hydraulically isolated to various degrees by interbedded fine-grained confining units" Tauchen et al. (2012).
Mzu (Mesozoic rocks undivided) (confining unit exists)	Clastic sedimentary rock aquifer (consolidated or semi-consolidated rock)	"The early Mesozoic Era was a time of shallow seas with deposition of interbedded layers (in decreasing abundance) of sandstone, siltstone, shale, carbonates, and evaporates . An emergent transition to a terrestrial environment during the Late Triassic and Early Jurassic Epochs resulted in the deposition of marginal marine, eolian, fluvial, and paludal sandstones and shales ." Tauchen et al. (2012) "The Paleozoic aquifer system comprises Permian through Ordovician carbonate and sandstone hydrogeologic units." Tauchen et al. (2012) "Immediately below the Cody [Major] confining unit, the lower and middle Mesozoic aquifers and confining units system comprises Cretaceous through Jurassic hydrogeologic units." Tauchen et al. (2012).
Pzu (Paleozoic rocks undivided)	Carbonate aquifer	"Paleozoic strata in the WBRB were deposited in marine and nonmarine transgressive/regressive environments. Marine limestones and dolomites are the dominant lithologies of the Paleozoic sequence, with less extensive sandstones and shales that represent beach and near-shore environments." Tauchen et al. (2012) "The Paleozoic aquifer system comprises Permian through Ordovician carbonate and sandstone hydrogeologic units." Tauchen et al. (2012) "Immediately below the Cody [Major] confining unit, the lower and middle Mesozoic aquifers and confining units system comprises Cretaceous through Jurassic hydrogeologic units." Tauchen et al. (2012).

Tauchen, P., Bartos, T.T., Clarey, K.E., Quillinan, S.A., Hallberg, L.L., Clark, M.L., Thompson, M., Gribb, N., Worman, B., Gracias, T. (2012). Wind/Bighorn River Basin Water Plan Update Groundwater Study Level 1 (2008–2011). Groundwater Determination. Wyoming Water Development Commission Technical Memorandum, 397 pp. Accessed June 5, 2022 from <https://waterplan.state.wy.us/plan/bighorn/2010/gw-finalrept/gw-finalrept.pdf>

3.49 Black Hills Uplift

Supplementary Fig. 194. Hydrogeologic cross section. 20 equally spaced transparent pink bars overlie the cross section; each shaded bar depicts the vertical offset from the land surface to the top of the uppermost confining unit or endogenous bedrock.

Supplementary Fig. 195. Vertical variations in the prevalence of wells that have been defined as tapping an unconfined or a confined aquifer by the USGS. The smaller squares represent 10 m depth intervals from the land surface to 100 m; the larger squares represent 20 m intervals from 100 m to 300 m below the land surface.

The Black Hills Uplift is located in western South Dakota.

(i) A hydrogeologic cross section presented in Fig. 25 by Driscoll et al. (2002) depicts a series of sedimentary rock units dipping to the east, with numerous confining units including the Spearfish confining unit and the Opeche confining unit.

(ii) We analysed wells within the study area that the USGS has defined as either unconfined or confined. Most (>80%) wells at depths of 180-200 m and at depths exceeding 180 m are defined as tapping a confined aquifer.

Depth to confined conditions: 180-200 m (see (ii) above)

Reference: Driscoll, D. G., Carter, J. M., Williamson, J. E., Putnam, L. D. (2002). Hydrology of the Black Hills area, South Dakota. US Geological Survey Water-Resources Investigations Report 2002-4094, 158 pp. Accessed February 16, 2021 from <https://pubs.usgs.gov/wri/wri024094/pdf/wri024094.pdf>

The table below presents a series of published quotes (see quotation marks denoting text quoted from another publication, which is cited following the quotation marks with the full reference written in full below the table). The leftmost column lists a title of a hydrogeologic formation depicted in the cross section on the previous page. The rightmost column presents a quote from a hydrogeological study (see base of table for citation). The quote has been annotated with colored text to highlight how we categorized each layer (i.e., see categories in the center column in the table). Specifically: (i) blue text highlights portions of a quote that provide insights into the degree of consolidation of the formation, (ii) red text highlights portions of a quote that categorize the formation as an aquifer or an aquitard (i.e., higher versus lower permeability in the context of local hydrogeologic formations), and (iii) green text highlights portions of a quote that provide information about the lithology of the formation.

Supplementary Table 52. Hydrostratigraphy details for the Black Hill Uplift

Formation name	Category	Quote
Unconsolidated units	Unconsolidated aquifer	“Unconsolidated deposits of Tertiary or Quaternary age, including alluvium, colluvium, and wind-blown deposits , all have the potential to be local aquifers where they are saturated. ” (Driscoll et al., 2002).
Cretaceous sequence confining unit (this unit is predominantly aquitard)	Sedimentary rock aquitard (consolidated or semi-consolidated rock)	“The Cretaceous-sequence confining unit mainly includes shales of low permeability, such as the Pierre Shale ; local aquifers in the Pierre Shale are referred to as the Pierre aquifer in this report.” (Driscoll et al., 2002).
Inyan Kara aquifer	Clastic sedimentary rock aquifer (consolidated or semi-consolidated rock)	“The Cretaceous-age Inyan Kara Group consists of the Lakota Formation and overlying Fall River Formation . The Lakota Formation consists of the Chilson, Minnewaste Limestone, and Fuson Shale members . The Lakota Formation consists of yellow, brown, and reddish-brown massive to thinly bedded sandstone, pebble conglomerate, siltstone, and claystone of fluvial origin (Gott and others, 1974); locally there are lenses of limestone and coal . The Fall River Formation is a brown to reddish-brown, fine-grained sandstone, thin bedded at the top and massive at the bottom (Strobel and others, 1999).” (Driscoll et al., 2002). “The Inyan Kara aquifer generally has the highest effective porosity of the major aquifers.” (Driscoll et al., 2002).
Jurassic-sequence semi-confining unit (this unit is predominantly aquitard)	Sedimentary rock aquitard (consolidated or semi-consolidated rock)	“The Jurassic-sequence semiconfining unit consists of shales and sandstones . Overall, this unit is semiconfining because of the low permeability of the interbedded shales ” (Driscoll et al., 2002).
Spearfish confining unit (this unit is predominantly aquitard)	Sedimentary rock aquitard (consolidated or semi-consolidated rock)	“Red silty shale, soft red sandstone and siltstone with gypsum and thin limestone layers. Gypsum locally near the base.” (Driscoll et al., 2002). “Spearfish confining unit” (Driscoll et al., 2002).
Minnekahta aquifer	Carbonate aquifer	“The Permian-age Minnekahta Limestone is a fine-grained, purple to gray laminated limestone (Strobel and others, 1999), which ranges in thickness from 25 to 65 ft in the study area. ” (Driscoll et al., 2002). “Transmissivity and hydraulic conductivity

Formation name	Category	Quote
		also may be high in the Minnelusa aquifer. " (Driscoll et al., 2002).
Opeche confining unit	Sedimentary rock aquitard (consolidated or semi-consolidated rock)	"Red shale and sandstone." (Driscoll et al., 2002). "Opeche confining unit" (Driscoll et al., 2002).
Minnelusa aquifer	Clastic sedimentary rock aquifer (consolidated or semi-consolidated rock)	"The Pennsylvanian- and Permian-age Minnelusa Formation consists mostly of yellow to red crossstratified sandstone, limestone, dolomite, and shale (Strobel and others, 1999)." (Driscoll et al., 2002). "The Pennsylvanian (or Minnelusa) aquifer is contained within the sandstones and limestones of the Minnelusa Formation, Tensleep Sandstone, Amsden Formation, and equivalents of Pennsylvanian age (fig. 12)." (Driscoll et al., 2002).
Madison aquifer	Carbonate aquifer	The Mississippian (or Madison) aquifer is contained within the limestones, siltstones, sandstones, and dolomite of the Madison Limestone or Group. Generally, water in the Mississippian aquifer is confined except in outcrop areas." (Driscoll et al., 2002).
Deadwood aquifer	Clastic sedimentary rock aquifer (consolidated or semi-consolidated rock)	"The Cambrian-Ordovician (or Deadwood) aquifer is contained within the sandstones of Cambrian age (Deadwood Formation and equivalents) and limestones of Ordovician age (Red River Formation and equivalents) (fig. 12)." (Driscoll et al., 2002).
Precambrian igneous and metamorphic units	Endogenous bedrock	"Schist, slate, quartzite, and arkosic grit. Intruded by diorite, metamorphosed to amphibolite, and by granite and pegmatite." (Driscoll et al., 2002).

Driscoll, D. G., Carter, J. M., Williamson, J. E., Putnam, L. D. (2002). Hydrology of the Black Hills area, South Dakota. US Geological Survey Water-Resources Investigations Report 2002-4094, 158 pp. Accessed May 31, 2022 from <https://pubs.usgs.gov/wri/wri024094/pdf/wri024094.pdf>

3.50 Black Warrior River Aquifer System (Eutaw and McShan Formations and Tuscaloosa Group)

Supplementary Fig. 196. Hydrogeologic cross section. 20 equally spaced transparent pink bars overlie the cross section; each shaded bar depicts the vertical offset from the land surface to the top of the uppermost confining unit or endogenous bedrock.

Supplementary Fig. 197. Vertical variations in the prevalence of wells that have been defined as tapping an unconfined or a confined aquifer by the USGS. The smaller squares represent 10 m depth intervals from the land surface to 100 m; the larger squares represent 20 m intervals from 100 m to 300 m below the land surface.

The Black Warrior River Aquifer System is located in eastern Mississippi and western Alabama.

(i) A hydrogeologic cross section presented in Fig. 3 by Strom and Mallory (1995) suggests that the shallowest confining unit is typically 3.5 m below the land surface, but with wide variability along the cross section (25th-75th percentile range of the depth to uppermost confining unit is 0m to 255 m).

(ii) We analysed wells within the study area that the USGS has defined as either unconfined or confined. Most (>80%) wells at depths of 0-10 m and at depths exceeding 0 m are defined as tapping a confined aquifer.

Depth to confined conditions: 0-10 m (based on (ii) above)

References: Strom, E.W., Mallory, M.J. (1995). Hydrogeology and simulation of ground-water flow in the Eutaw-McShan Aquifer and in the Tuscaloosa aquifer system in northeastern Mississippi. US Geological Survey Water-Resources Investigations Report 94-4223, 89 pp. Accessed November 29, 2021 from <https://pubs.usgs.gov/wri/1994/4223/report.pdf>

The table below presents a series of published quotes (see quotation marks denoting text quoted from another publication, which is cited following the quotation marks with the full reference written in full below the table). The leftmost column lists a title of a hydrogeologic formation depicted in the cross section on the previous page. The rightmost column presents a quote from a hydrogeological study (see base of table for citation). The quote has been annotated with colored text to highlight how we categorized each layer (i.e., see categories in the center column in the table). Specifically: (i) blue text highlights portions of a quote that provide insights into the degree of consolidation of the formation, (ii) red text highlights portions of a quote that categorize the formation as an aquifer or an aquitard (i.e., higher versus lower permeability in the context of local hydrogeologic formations), and (iii) green text highlights portions of a quote that provide information about the lithology of the formation.

Supplementary Table 53. Hydrostratigraphy details for the Black Warrior River Aquifer System

Formation name	Category	Quote
Midway Group	Sedimentary rock aquitard (consolidated or semi-consolidated rock)	"South and west of the Ripley Cuesta is another belt of lowlands, the Flatwoods, underlain by the Paleocene Porters Creek Clay of the Midway Group." Mallory (1993) "The Midway Group is composed predominantly of marine clay and shale but includes subordinate sand and limestone beds. " Mallory (1993)
Ripley aquifer (part of Selma Group)	Clastic sedimentary rock aquifer (consolidated or semi-consolidated rock)	"An important aquifer in Tippah, eastern Benton, and northern Union Counties. The McNoiry sand member ranges up to 250 feet thick and is the unit which most of the wells utilize in the Ripley. Quality is good except for hardness." (Shows, 1970) "The Ripley Formation in Alabama and Mississippi has a maximum thickness of about 500 ft. The formation typically consists of clay, sandy clay, sand, and thin beds of sandstone." Mallory (1993)
Demopolis Chalk and Mooreville Chalk (part of Selma Group; "Coffee sand" formation may be an aquifer in areas)	Sedimentary rock aquitard (consolidated or semi-consolidated rock)	"These formations in western Alabama and eastern Mississippi, in ascending order, are the Mooreville Chalk, the Demopolis Chalk, the Ripley Formation, and the Prairie Bluff Chalk. " Cushing et al., (1964) "Not an aquifer." (referring to Demopolis Chalk and Mooreville chalk) (Shows, 1970)
Tombigbee Sand Member	Clastic sedimentary rock aquifer (consolidated or semi-consolidated rock)	"A persistent sand at the top of the formation, known as the Tombigbee Sand Member (Hilgard, 1860, p. 61), is massive, highly glauconitic, calcareous, and fossiliferous in the upper part." Cushing et al., (1964) "An important aquifer in most of Tishomingo and all of Alcorn and Prentiss Counties." In Table 3, (Shows, 1970).
Eutaw Formation	Clastic sedimentary rock aquifer (consolidated or semi-consolidated rock)	"The main body of the formation is composed of gray clay interbedded with fine glauconitic sand. Thin beds of fine to medium glauconitic sand are common and are fairly persistent near the base of the formation, which is normally marked by a thin bed of fine gravel. The sands are commonly cross bedded or show distinct stratification. " Cushing et al., (1964) "Thin beds of fine to medium glauconitic sand within the Eutaw and McShan Formations make up the bulk of the upper Black Warrior River regional aquifer, locally known as the Eutaw-McShan aquifer. " Mallory (1993)
McShan Formation	Clastic sedimentary rock aquifer (consolidated or	"It consists of laminated micaceous glauconitic gray clay, fine sand, and lenticular beds of fine to medium glauconitic sand. " Cushing et al., (1964) "Thin beds of fine to medium

Formation name	Category	Quote
	semi-consolidated rock	glauconitic sand within the Eutaw and McShan Formations make up the bulk of the upper Black Warrior River regional aquifer, locally know as the Eutaw-McShan aquifer." Mallory (1993)
Gordo Formation	Clastic sedimentary rock aquifer (consolidated or semi-consolidated rock)	"It is composed of thick beds of sand containing gravel in the lower part and multicolored clay and shale interbedded with sand in the upper part." Cushing et al., (1964) "The Gordo Formation is an important aquifer in Alabama and Mississippi." Cushing et al., (1964)
Coker Formation	Clastic sedimentary rock aquifer (consolidated or semi-consolidated rock)	"In Alabama, the Coker has been subdivided into the Eoline Member and an upper unnamed member." Cushing et al., (1964) "The Eoline Member (Monroe and others, 1946, p. 194-197) consists of thin-bedded clay, sandy clay, shale, and sand, mostly of marine origin; subordinate beds of sand occur throughout the unit." Cushing et al., (1964) "Permeable sand and gravel beds in the Coker and Gordo Formations of the Tuscaloosa Group make up the lower Black Warrior River aquifer." Mallory (1993)
Massive sand	Clastic sedimentary rock aquifer (consolidated or semi-consolidated rock)	"The massive sand is "a series of medium- to coarse grained sands * * *" (according to McGlothlin, 1944, p. 40). Interbedded shale and clay occur in the thick beds of coarse sand, chert, and quartz gravel which compose the main body of the unit." Cushing et al., (1964) "Although the massive sand is not generally used as a source of ground water, it is potentially one of the most important aquifers in the embayment." Mallory (1993)
Lower Cretaceous rocks	Clastic sedimentary rock aquifer (consolidated or semi-consolidated rock)	"Lower Cretaceous rocks do not crop out on the eastern side of the embayment. However, in Mississippi and Alabama they occur in the subsurface as thick sands, clays, and shales." Cushing et al., (1964) "In places, some of the uppermost beds of the Lower Cretaceous series are lithologically very similar to the basal beds of the over-lying Coker Formation and are included in this aquifer."
Paleozoic rocks	Sedimentary rock aquitard (consolidated or semi-consolidated rock)	"The lower Black Warrior River aquifer is the lowest aquifer in the Southeastern Coastal Plain aquifer system. In the northern part of the study area, the basal confining unit of the Southeastern Coastal Plain aquifer system consists of Paleozoic rocks. These consolidated shales, sandstones, limestones, and dolomites have much smaller permeability than the overlying Cretaceous sediments." Mallory (1993)

Cushing, E. M., Boswell, E. H., Hosman, R. L. (1964). General geology of the Mississippi Embayment. US Geological Survey Professional Paper 448-B, 32 pp. Accessed February 21, 2022 from <https://pubs.usgs.gov/pp/0448b/report.pdf>

Mallory, M. J. (1993). Hydrogeology of the Southeastern Coastal Plain Aquifer System in Parts of Eastern Mississippi and Western Alabama. Regional aquifer-system analysis. Southeastern coastal plain. US Geological Survey Professional Paper 1410-G, 66 pp. Accessed February 21, 2022 from <https://pubs.usgs.gov/pp/1410g/report.pdf>

Shows, T.N. (1970). Water resources of Mississippi. Mississippi Geological, Economic and Topographical Survey Bulletin 113, 182 pp. Accessed February 21, 2022 from <https://www.mdeq.ms.gov/wp-content/uploads/2017/06/Bulletin-113.pdf>

3.51 Castle Hayne Aquifer

Supplementary Fig. 198. Hydrogeologic cross section. 20 equally spaced transparent pink bars overlie the cross section; each shaded bar depicts the vertical offset from the land surface to the top of the uppermost confining unit or endogenous bedrock.

Supplementary Fig. 199. Vertical variations in the prevalence of wells that have been defined as tapping an unconfined or a confined aquifer by the USGS. The smaller squares represent 10 m depth intervals from the land surface to 100 m; the larger squares represent 20 m intervals from 100 m to 300 m below the land surface.

The Castle Hayne Aquifer System is located in the eastern portion of North Carolina.

(i) A hydrogeologic cross section presented in Plate 6 by Winner Jr. and Coble (1989) depicts shallow low-permeability units including the Castle Hayne Confining Unit, Beaufort Confining Unit, and Peedee Confining Unit.

(ii) We analysed wells within the study area that the USGS has defined as either unconfined or confined. Most (>80%) wells at depths of 10-20 m and at depths exceeding 10 m are defined as tapping a confined aquifer.

Depth to confined conditions: 10-20 m (based on (ii) above)

Reference: Winner Jr., M.D., Coble, R.W. (1989). Hydrogeologic framework of the North Carolina Coastal Plain aquifer system. US Geological Survey Report 87-690. 167 pp. Accessed April 1, 2021 from <https://pubs.usgs.gov/of/1987/0690/report.pdf>

The table below presents a series of published quotes (see quotation marks denoting text quoted from another publication, which is cited following the quotation marks with the full reference written in full below the table). The leftmost column lists a title of a hydrogeologic formation depicted in the cross section on the previous page. The rightmost column presents a quote from a hydrogeological study (see base of table for citation). The quote has been annotated with colored text to highlight how we categorized each layer (i.e., see categories in the center column in the table). Specifically: (i) blue text highlights portions of a quote that provide insights into the degree of consolidation of the formation, (ii) red text highlights portions of a quote that categorize the formation as an aquifer or an aquitard (i.e., higher versus lower permeability in the context of local hydrogeologic formations), and (iii) green text highlights portions of a quote that provide information about the lithology of the formation.

Supplementary Table 54. Hydrostratigraphy details for the Castle Hayne Aquifer

Formation name	Category	Quote
Surficial	Unconsolidated aquifer	"The surficial aquifer (A10) overlies all of the North Carolina Coastal Plain (fig. 1) and consists of fine sand, silt, clay, shell, and peat beds . Scattered deposits of coarser-grained sediments in the unit occur in relict beach ridges or in alluvium. " (Giese et al., 1997)
Yorktown Confining unit	Sedimentary rock aquitard (consolidated or semi-consolidated rock)	"The Yorktown confining unit (CU9) overlying the Yorktown aquifer is comprised of the youngest clay beds of the Yorktown Formation in most places, but locally may include clay beds of Pleistocene or Holocene age. Its thickness averages about 25 ft , ranging from less than 10 up to 50 ft thick. It is composed largely of clay and sandy clay that locally includes beds of fine sand or shell."
Yorktown	Clastic sedimentary rock aquifer (consolidated or semi-consolidated rock)	"The Yorktown aquifer largely consists of fine sand, silty and clayey sand, sand with shells and shell beds, some limestone, and some coarse sand beds. " (Giese et al., 1997).
Pungo River Confining unit	Sedimentary rock aquitard (consolidated or semi-consolidated rock)	"The Pungo River confining unit (CU8) is formed by the upper clay beds of the Pungo River Formation and contiguous clays of the lowermost Yorktown Formation . The confining unit ranges in thickness from less than 10 ft near the western margin to about 150 ft beneath Currituck County, with an average thickness of nearly 55 ft." (Giese et al., 1997)
Pungo River	Clastic sedimentary rock aquifer (consolidated or semi-consolidated rock)	"The Pungo River aquifer is composed of fine to medium marine sands having considerable phosphate content." (Giese et al., 1997). "The Pungo River aquifer (A8) is thinnest near its western and northern limits, where its thickness averages about 15 ft . The aquifer dips eastward and thickens to more than 200 ft in the vicinity of the Outer Banks, where the top is deeper than 700 ft below sea level. " (Giese et al., 1997)
Castle Hayne Confining unit	Sedimentary rock aquitard (consolidated or semi-consolidated rock)	"The thickness of the Castle Hayne confining unit (CU7) averages only about 10 ft; it exceeds 25 ft only in Gates County along the Virginia border, in eastern Pamlico and Carteret Counties, and in two small areas along the western limit of the Castle Hayne aquifer (A7). The confining unit is composed of beds of clay, sandy clay, and clay with sandy streaks that are part of the Pungo River Formation, the Yorktown Formation, or younger clays. " (Giese et al., 1997)

Formation name	Category	Quote
Castle Hayne	Carbonate aquifer	"The Castle Hayne aquifer (A7) consists of limestone, sand, and minor amounts of clay deposited under marine conditions . Limestone may occur as shell limestone, dolomitic limestone, and sandy limestone ranging from loosely consolidated to hard and recrystallized ." (Giese et al., 1997).
Beaufort Confining unit	Sedimentary rock aquitard (consolidated or semi-consolidated rock)	"The Beaufort confining unit (CU6) consists of the uppermost sediments of the Beaufort Formation and possibly some younger clay, silt, and sandy clay . Over most of the area, the confining unit shows a gradation from sandy clay to clay, but contains distinct clay beds interlayered with fine sand or silt ." (Giese et al., 1997)
Beaufort aquifer	Clastic sedimentary rock aquifer (consolidated or semi-consolidated rock)	"The Beaufort aquifer (A6) consists of fine to medium glauconitic sands, clayey sands, and clay beds of marine origin . Shell and limestone beds are present but are less than 6 ft thick." (Giese et al., 1997).
Peedee Confining unit	Sedimentary rock aquitard (consolidated or semi-consolidated rock)	"The Peedee confining unit (CU5) is composed of clay, silty clay, and sandy clay . Winner and Coble (1989) did not identify the confining unit with a particular geologic unit, but the unit is composed primarily of sediments at the Cenozoic-Mesozoic boundary ." (Giese et al., 1997)
Peedee aquifer	Clastic sedimentary rock aquifer (consolidated or semi-consolidated rock)	"The Peedee aquifer (A5) consists of fine to medium sands interbedded with clays and silts . Thin beds of consolidated calcareous sandstone and impure limestone are interlayered among the sands in some places , particularly in the southeastern North Carolina Coastal Plain area." (Giese et al., 1997).
Black Creek Confining unit	Sedimentary rock aquitard (consolidated or semi-consolidated rock)	"The Black Creek confining unit (CU4) is primarily composed of the uppermost beds of the Black Creek Formation and consists of clay, silty clay, and sandy clay ." (Giese et al., 1997).
Black Creek aquifer	Clastic sedimentary rock aquifer (consolidated or semi-consolidated rock)	"The Black Creek aquifer (A4) contains Upper Cretaceous sediments of both the Black Creek and underlying Middendorf Formations (Winner and Coble, 1989, 1996). The Black Creek Formation consists mainly of thinly laminated gray to black clay, interbedded with gray to tan sands . Outcrops also exhibit sand- or clay-dominated lenses ." (Giese et al., 1997)
Upper Cape Fear Creek Confining unit	Sedimentary rock aquitard (consolidated or semi-consolidated rock)	"the upper Cape Fear confining unit (CU3) consists of nearly continuous clay, silty clay, and sandy clay beds belonging either to the Middendorf Formation in the Sand Hills area or to the Black Creek Formation. The thickness of the confining unit averages about 48 ft (Winner and Coble, 1989) but may exceed 100 ft " (Giese et al., 1997)
Upper Cape Fear aquifer	Clastic sedimentary rock aquifer (consolidated or semi-consolidated rock)	"The upper Cape Fear aquifer (A3) varies in thickness from about 10 ft along its western edge to nearly 500 ft in central Tyrrell County. The average thickness of the aquifer is slightly more than 100 ft." (Giese et al., 1997). "The Cape Fear aquifer consists predominantly of sand, silt, and gravel separated by relatively thick silt and clay layers ." (Aucott, 1996).

Formation name	Category	Quote
Lower Cape Fear Confining unit	Sedimentary rock aquitard (consolidated or semi-consolidated rock)	"The lower Cape Fear confining unit (CU2) is composed of clay and sandy-clay beds that belong largely to the Cape Fear Formation. The average thickness of the confining unit is about 50 ft." (Giese et al., 1997)
Lower Cape Fear aquifer	Clastic sedimentary rock aquifer (consolidated or semi-consolidated rock)	"The lower Cape Fear aquifer (A2) strikes northeast and dips southwest at a rate of 15 to 35 ft/mi. Its extent is shown in figure 20. Its thickness ranges from a few feet along its western margin to more than 400 ft in the north-eastern North Carolina Coastal Plain." (Giese et al., 1997). "The Cape Fear aquifer consists predominantly of sand, silt, and gravel separated by relatively thick silt and clay layers." (Aucott, 1996).
Lower Cretaceous aquifer	Clastic sedimentary rock aquifer (consolidated or semi-consolidated rock)	"Various investigators have established that the updip beds of the Lower Cretaceous aquifer are largely nonmarine in origin, but the incidence of beds of marine origin increases downdip toward the coast. The non-marine beds are shales, sands, and gravel. Marine beds are chiefly limestones that may be sandy or dolomitic." (Giese et al., 1997).
Crystalline Basement rock	Endogenous bedrock	"The Lower Cretaceous aquifer and confining unit are overlain everywhere by the lower Cape Fear aquifer (A2) and are underlain everywhere by crystalline basement rocks" . (Giese et al., 1997). "The unconsolidated Coastal Plain aquifer system is underlain by crystalline basement rocks of low permeability." (Giese et al., 1991).

Kennedy, C.D., Genereux, D.P. (2007). ¹⁴C groundwater age and the importance of chemical fluxes across aquifer boundaries in confined Cretaceous aquifers of North Carolina, USA. Radiocarbon, 49, 1181-1203.

Giese, G.L., Eimers, J.L., Coble, R.W. (1997). Simulation of ground water flow in the Coastal Plain aquifer system of North Carolina. US Geological Survey Professional Paper 1404-M. 142 p. accessed on 3/29/2022 via <https://pubs.usgs.gov/pp/1404m/report.pdf>

Aucott, W.R. (1996). Hydrology of the Southeastern Coastal Plain aquifer system in South Carolina and parts of Georgia and North Carolina (No. 1410-E). US Geological Survey. <https://pubs.er.usgs.gov/publication/pp1410E>

Giese, G.L., Eimers, J.L., Coble, R.W. (1991). Simulation of ground-water flow in the Coastal Plain aquifer system of North Carolina (Vol. 1404). US Government Printing Office. Accessed on March 29, 2022 via <https://pubs.er.usgs.gov/publication/ofr90372>

3.52 Coachella Valley

Supplementary Fig. 200. Hydrogeologic cross section. 20 equally spaced transparent pink bars overlie the cross section; each shaded bar depicts the vertical offset from the land surface to the top of the uppermost confining unit or endogenous bedrock.

Supplementary Fig. 201. Vertical variations in the prevalence of wells that have been defined as tapping an unconfined or a confined aquifer by the USGS. The smaller squares represent 10 m depth intervals from the land surface to 100 m; the larger squares represent 20 m intervals from 100 m to 300 m below the land surface.

Coachella Valley is located in southcentral California.

(i) A hydrogeologic cross section presented in Fig. 3-3 by MWH (2014) suggests that a relatively continuous aquitard exists in the southeastern portion of the Coachella Valley, but that this unit is absent in the northwestern portion of the area.

(ii) We analysed wells within the study area that the USGS has defined as either unconfined or confined. Nearly all (11 of 12) wells in the Coachella Valley are defined as unconfined, including the deepest wells in the dataset (depths of 323 m and 341 m).

Depth to confined conditions: >341 m (based on (ii) above)

Reference: MWH (2014). Technical memorandum No.1 preliminary data review and documentation of technical methods. Technical memorandum submitted to Coachella Valley Salt and Nutrient Management Plan Technical Group, 122 pp. Accessed April 11, 2022 via <http://cvwd.org/DocumentCenter/View/1158/Final-TM-No-1-Preliminary-Data-Review-and-Documentation-of-Technical-Methods-PDF?bidl=>

The table below presents a series of published quotes (see quotation marks denoting text quoted from another publication, which is cited following the quotation marks with the full reference written in full below the table). The leftmost column lists a title of a hydrogeologic formation depicted in the cross section on the previous page. The rightmost column presents a quote from a hydrogeological study (see base of table for citation). The quote has been annotated with colored text to highlight how we categorized each layer (i.e., see categories in the center column in the table). Specifically: (i) blue text highlights portions of a quote that provide insights into the degree of consolidation of the formation, (ii) red text highlights portions of a quote that categorize the formation as an aquifer or an aquitard (i.e., higher versus lower permeability in the context of local hydrogeologic formations), and (iii) green text highlights portions of a quote that provide information about the lithology of the formation.

Supplementary Table 55. Hydrostratigraphy details for the Coachella Valley

Formation name	Category	Quote
Semi-perched zone	Unconsolidated aquifer	"The semi-perched aquifer is characterized by fine-grained Holocene and Recent lake deposits and alluvium that form an effective barrier to the deep percolation of surface runoff and applied water within the central portion of the East Valley where present." MWH (2014). "The semi-perched aquifer consists of interbedded layers of fine sand and clay and is separated from the underlying upper aquifer by a laterally discontinuous clay zone (DWR, 1964)." MWH (2014).
Upper aquifer	Unconsolidated aquifer	"Based on DWR (1964), the upper aquifer , which is formed of Upper Pleistocene alluvium , underlies the semi-perched aquifer. The upper aquifer typically consists of coarse sand and gravel with discontinuous clay lenses in the West Valley and the northern part of the East Valley. Finer sand and sandy clay dominate in the southern part of the East Valley. The upper aquifer is believed to be unconfined or semi-confined in most of the West Valley, and is confined in most of the East Valley by the semi-perched aquifer and a discontinuous clay layer (referred to as the aquitard)." MWH (2014).
Aquitard	Sedimentary rock aquitard (consolidated or semi-consolidated rock)	"The aquitard typically consists of clay and sandy clay with discontinuous sand lenses having low permeability. " MWH (2014).
Lower aquifer	Unconsolidated aquifer	The lower aquifer zone , composed in part of the Ocotillo conglomerate , consists of silty sands and gravels with interbeds of silt and clay. It is the most important source of groundwater in the Whitewater River Subbasin." MWH (2014).
Bedrock	Endogenous bedrock	"DWR (1964) inferred the depth to bedrock was in excess of 12,000 feet below ground surface based on gravity survey data." MWH (2014). " basement bedrock " MWH (2014).

MWH (2014). Technical memorandum No.1 preliminary data review and documentation of technical methods. Technical memorandum submitted to Coachella Valley Salt and Nutrient Management Plan Technical Group, 122 pp. Accessed June 2, 2022 via <http://cvwd.org/DocumentCenter/View/1158/Final-TM-No-1-Preliminary-Data-Review-and-Documentation-of-Technical-Methods-PDF?bidId=>

3.53 Cuyama Valley

Supplementary Fig. 202. Hydrogeologic cross section. 20 equally spaced transparent pink bars overlie the cross section; each shaded bar depicts the vertical offset from the land surface to the top of the uppermost confining unit or endogenous bedrock.

Supplementary Fig. 203. Vertical variations in the prevalence of wells that have been defined as tapping an unconfined or a confined aquifer by the USGS. The smaller squares represent 10 m depth intervals from the land surface to 100 m; the larger squares represent 20 m intervals from 100 m to 300 m below the land surface.

The Cuyama Valley is located in western California.

(i) A hydrogeologic cross section presented in Fig. 2 by Sweetkind et al. (2013) does not depict a clear confining unit

(see cross section to the left). The median depth to a confined unit or endogenous bedrock is >722 m.

(ii) We analysed wells within the study area that the USGS has defined as either unconfined or confined. The available USGS well data (n=2 wells with depths of 71 m and 183 m) are insufficient to evaluate the depths at which the valley transitions from unconfined to confined conditions.

(iii) In their 2021 Groundwater Sustainability Plan, Woodard and Curran report (direct quote): “The aquifer is considered to be continuous and unconfined with the exception of locally perched aquifers resulting from clays in the formations.”

Depth to confined conditions: >722 m based on (i) and (iii)

References: Woodard and Curran (2021). Cuyama Basin Groundwater Sustainability Plan, Chapter 2, Basin Settings, 168 pp. Accessed May 16, 2022 via <https://cuyamabasin.org/assets/pdf/public-final-gsp/Cuyama-Final-GSP-Chapter-2.pdf>

Sweetkind, D.S., Faut, C.C., Hanson, R.T. (2013). Construction of 3-D geologic framework and textural models for Cuyama Valley groundwater basin, California. US Geological Survey Scientific Investigations Report 2013-5127, 58 pp. Accessed March 7, 2021 from <https://pubs.usgs.gov/sir/2013/5127/pdf/sir2013-5127.pdf>

The table below presents a series of published quotes (see quotation marks denoting text quoted from another publication, which is cited following the quotation marks with the full reference written in full below the table). The leftmost column lists a title of a hydrogeologic formation depicted in the cross section on the previous page. The rightmost column presents a quote from a hydrogeological study (see base of table for citation). The quote has been annotated with colored text to highlight how we categorized each layer (i.e., see categories in the center column in the table). Specifically: (i) blue text highlights portions of a quote that provide insights into the degree of consolidation of the formation, (ii) red text highlights portions of a quote that categorize the formation as an aquifer or an aquitard (i.e., higher versus lower permeability in the context of local hydrogeologic formations), and (iii) green text highlights portions of a quote that provide information about the lithology of the formation.

Supplementary Table 56. Hydrostratigraphy details for the Cuyama Valley

Formation name	Category	Quote
Alluvial aquifers	Unconsolidated aquifer	"Geologic maps of the Cuyama Valley (Vedder and Repenning, 1975; Kellogg and others, 2008) show two units of Holocene and Pleistocene-aged alluvial deposits , termed younger and older alluvium, underlying the three monitoring well sites." Everett et al., 2013. "The main water-bearing deposits in the study area are the saturated portions of the younger and older alluvium and the Morales Formation." (Everett et al. 2013)
Morales formation aquifer	Clastic sedimentary rock aquifer (consolidated or semi-consolidated rock)	"The Morales Formation (QTm, fig. 2) is a Pliocene Pleistocene fluvial deposit that is up to 5,000 feet thick and consists of massive- to thick-bedded, partly consolidated deposits of clay, silt, sand, and gravel (Hill and others, 1958; Ellis and others, 1993)." Everett et al., 2013. "The main water-bearing deposits in the study area are the saturated portions of the younger and older alluvium and the Morales Formation." (Everett et al. 2013)
Bedrock	Sedimentary rock aquitard (consolidated or semi-consolidated rock) OR Endogenous bedrock (*for display purposes, we show a sedimentary rock aquitard in Fig. 1)	" non-water-bearing rocks—the crystalline granitic rocks and all consolidated sedimentary rocks older than the Morales Formation". Everett et al., 2013. "All rocks that are older than the Morales Formation were considered by previous investigators to be non-water-bearing " (Everett et al., 2013)

Everett, R. R., Gibbs, D. R., Hanson, R. T., Sweetkind, D. S., Brandt, J. T., Falk, S. E., Harich, C. R. (2013). Geology, water-quality, hydrology, and geomechanics of the Cuyama Valley groundwater basin, California, 2008-12. US Geological Survey Scientific Investigations Report 2013-5108, 76 pp. Accessed February 21, 2022 from <https://pubs.usgs.gov/sir/2013/5108/pdf/sir2013-5108.pdf>

3.54 Denver Basin

Supplementary Fig. 204. Hydrogeologic cross section. 20 equally spaced transparent pink bars overlie the cross section; each shaded bar depicts the vertical offset from the land surface to the top of the uppermost confining unit or endogenous bedrock.

Supplementary Fig. 205. Vertical variations in the prevalence of wells that have been defined as tapping an unconfined or a confined aquifer by the USGS. The smaller squares represent 10 m depth intervals from the land surface to 100 m; the larger squares represent 20 m intervals from 100 m to 300 m below the land surface.

The Denver Basin is located in the central United States.

(i) A hydrogeologic cross section presented in Fig. 3 by Malenda and Penn (2020) suggests the top of the uppermost confining

unit (the Laramie Formation) is typically 297 meters below the land surface (i.e. median of pink bars in cross section to left).

(ii) We analysed wells within the study area that the USGS has defined as either unconfined or confined. Most (>80%) wells at depths of 160-180 m and at depths exceeding 160 m are defined as tapping a confined aquifer.

(iii) Malenda and Penn (2020) highlight that confined conditions can exist in the Denver Formation, which overlies the Laramie Formation; they state that "The three Denver aquifer wells represent confined aquifer conditions". Therefore, we do not rely solely on the depth to the Laramie Formation to estimate the depth to confined conditions, instead drawing information from the wells where the USGS has defined whether the well taps unconfined versus confined conditions.

Depth to confined conditions: 160-180 m (based on (ii) above)

Reference: Malenda, H.F., Penn, C.A. (2020). Groundwater levels in the Denver Basin bedrock aquifers of Douglas County, Colorado, 2011–19. U.S. Geological Survey Scientific Investigations Report 2020–5076, 44 pp., Accessed November 29, 2021 from <https://pubs.usgs.gov/sir/2020/5076/sir20205076.pdf>

The table below presents a series of published quotes (see quotation marks denoting text quoted from another publication, which is cited following the quotation marks with the full reference written in full below the table). The leftmost column lists a title of a hydrogeologic formation depicted in the cross section on the previous page. The rightmost column presents a quote from a hydrogeological study (see base of table for citation). The quote has been annotated with colored text to highlight how we categorized each layer (i.e., see categories in the center column in the table). Specifically: (i) blue text highlights portions of a quote that provide insights into the degree of consolidation of the formation, (ii) red text highlights portions of a quote that categorize the formation as an aquifer or an aquitard (i.e., higher versus lower permeability in the context of local hydrogeologic formations), and (iii) green text highlights portions of a quote that provide information about the lithology of the formation.

Supplementary Table 57. Hydrostratigraphy details for the Denver Basin

Formation name	Category	Quote
Denver Formation	Clastic sedimentary rock aquifer (consolidated or semi-consolidated rock)	"Alluvial fan, swamp, overbank deposits; andesitic fluvial sandstone with volcanic ash deposits, coal, lignite, mudstone/claystone. Fe-rich sediments; sediments source of Se and U to groundwater." (Musgrove et al., 2014) "Confined to unconfined aquifer" (Musgrove et al., 2014). "Confined in central part. Contains a water table only near out-crops. Moderately permeable. May yield as much as 200 gallons per minute. " (Robson & Banta, 1995)
Arapahoe Formation	Clastic sedimentary rock aquifer (consolidated or semi-consolidated rock)	"Fluvial environment, alluvial fan deposits near mountain front; conglomerates, sandstone, siltstone, shale; pebbles and cobbles with granite, chert, metamorphic rocks, and quartz; shale more prevalent in northern part of basin." (Musgrove et al., 2014). "Productive confined aquifer" - (Musgrove et al., 2014).
Laramie Formation	Sedimentary rock aquitard (consolidated or semi-consolidated rock)	"Upper part shale, silty shale, silt stone, and interbedded fine sandstone." (Robson & Banta, 1995). "Shale is impermeable". (Robson & Banta, 1995). See figure 83 from Ref (Robson & Banta, 1995)
Laramie-Fox Hill Formation	Clastic sedimentary rock aquifer (consolidated or semi-consolidated rock)	"Laramie Formation: swamps, deltas, overbank deposits; claystone, coal, fluvial channel sandstone; contains coal and lignite beds. Fox Hills Sandstone: marine beach and delta-front environment; sandstone, thin siltstone and claystone beds; contains marine fossils." (Musgrove et al., 2014) "Productive confined to unconfined aquifer" – (Musgrove et al., 2014)
Pierre shale	Sedimentary rock aquitard (consolidated or semi-consolidated rock)	"A thick unit of low-permeability Cretaceous age Pierre Shale underlies the Laramie-Fox Hills aquifer and forms the base of the aquifer system" – (Musgrove et al., 2014)

Musgrove, M., Beck, J. A., Paschke, S. S., Bauch, N. J., Mashburn, S. L. (2014). Quality of Groundwater in the Denver Basin Aquifer System, Colorado, 2003-5. US Geological Survey Scientific Investigations Report 2014-5051, 123 pp. Accessed February 21, 2022 from <https://pubs.usgs.gov/sir/2014/5051/pdf/sir2014-5051.pdf>

Robson, S. G., Banta, E. R. (1995). Ground water atlas of the United States: Segment 2, Arizona, Colorado, New Mexico, Utah. US Geological Survey Hydrologic Investigations Atlas 730-C, Segment 2, 34 pp. Accessed February 21, 2022 from <https://pubs.usgs.gov/ha/730c/report.pdf>

3.55 Eastern Dakota Aquifer

Supplementary Fig. 206. Hydrogeologic cross section. 20 equally spaced transparent pink bars overlie the cross section; each shaded bar depicts the vertical offset from the land surface to the top of the uppermost confining unit or endogenous bedrock.

Supplementary Fig. 207. Vertical variations in the prevalence of wells that have been defined as tapping an unconfined or a confined aquifer by the USGS. The smaller squares represent 10 m depth intervals from the land surface to 100 m; the larger squares represent 20 m intervals from 100 m to 300 m below the land surface.

The Eastern Dakota Aquifer is located in northwest Iowa.

(i) A hydrogeologic cross section presented in the figure on page 27 by Prior et al. (2003) depicts relatively thick unconsolidated deposits overlying the Dakota Aquifer, which itself overlies dipping sedimentary sequences including several aquitards (e.g., the Upper and Basal Devonian Aquitards).

(ii) We analysed wells within the study area that the USGS has defined as either unconfined or confined. Most (>80%) wells at depths of 60-70 m and at depths exceeding 60 m are defined as tapping a confined aquifer.

Depth to confined conditions: 60-70 m (based on (ii) above)

Reference: Prior, J.C., Boekhoff, J.L., Howes, M.R., Libra, R.D., VanDorpe, P.E. (2003). Iowa's Groundwater Basics. Iowa Department of Natural Resources Report, 92 pp. Accessed November 29, 2021 from https://s-ihr34.ihr.uiowa.edu/publications/uploads/2014-08-24_08-08-21_es-06.pdf

The table below presents a series of published quotes (see quotation marks denoting text quoted from another publication, which is cited following the quotation marks with the full reference written in full below the table). The leftmost column lists a title of a hydrogeologic formation depicted in the cross section on the previous page. The rightmost column presents a quote from a hydrogeological study (see base of table for citation). The quote has been annotated with colored text to highlight how we categorized each layer (i.e., see categories in the center column in the table). Specifically: (i) blue text highlights portions of a quote that provide insights into the degree of consolidation of the formation, (ii) red text highlights portions of a quote that categorize the formation as an aquifer or an aquitard (i.e., higher versus lower permeability in the context of local hydrogeologic formations), and (iii) green text highlights portions of a quote that provide information about the lithology of the formation.

Supplementary Table 58. Hydrostratigraphy details for the Eastern Dakota Aquifer

Formation name	Category	Quote
Unconsolidated Quaternary deposits	Unconsolidated aquifer	From Prior et al., 2003 page 14: Hydrogeologic units: "Alluvium", "Glacial drift", and "Buried Valley". Dominant geologic material: "Sand, gravel, silt, clay", "Pebbly clay, silt, sand & gravel", and "Sand & gravel". Hydrologic condition: "Local to regional aquifers", "local sand & gravel aquifers", and "local to regional aquifers" respectively.
Cretaceous aquitard	Sedimentary rock aquitard (consolidated or semi-consolidated rock)	From Prior et al., 2003 page 14: Hydrogeologic units: "Cretaceous confining units" Dominant geologic material: "Shale, limestone". Hydrologic condition: "Confining beds; aquitard".
Cretaceous Aquifer (Dakota)	Clastic sedimentary rock aquifer (consolidated or semi-consolidated rock)	From Prior et al., 2003 page 14: Hydrogeologic units: "Dakota aquifer" Dominant geologic material: "Sandstone". Dominant geologic material: "Sandstone". Hydrologic condition: "Regional aquifer". "The two uppermost hydrogeologic units, the drift and the Cretaceous aquifers, generally are unconfined and are the top layer of the regional flow model (Mandie and Kontis, in press)." (Young, 1992).
Mississippian Aquifer	Carbonate aquifer	From Prior et al., 2003 page 14: Hydrogeologic units: "Mississippian aquifer". Dominant geologic material: "limestone, sandstone, shale", "dolomite, shale, limestone, chert", and "dolomite, limestone, chert". hydrologic condition: "local aquitard" & "regional aquifer". "The Mississippian aquifer is one of the most dependable sources of groundwater in north-central Iowa. Wells drilled into the Mississippian aquifer supply large volumes of water to livestock, industries, and municipalities." (Gannon & McKay, 2013).
Upper Devonian Shale aquitard (local aquifers exist)	Sedimentary rock aquitard (consolidated or semi-consolidated rock)	From Prior et al., 2003 page 14: Hydrogeologic units: "Devonian confining unit" Dominant geologic material: "shale, siltstone, dolomite". & "shale, dolomite, limestone". Hydrologic condition: "confining beds; aquitard", & "confining beds; aquitard, local aquifers".
Devonian aquifer	Carbonate aquifer	From Prior et al., 2003 page 14: Hydrogeologic units: "Devonian aquifer" Dominant geologic material: "dolomite, chert, limestone". Hydrologic condition: "regional aquifer".
Basal Devonian	Sedimentary rock aquitard (consolidated)	From Prior et al., 2003 page 14: Hydrogeologic units: "Devonian aquifer" Dominant geologic material:

Formation name	Category	Quote
Aquitard (evaporite, shale)	or semi-consolidated rock)	“limestone, dolomite, shale, gypsum” . Hydrologic condition: “local aquitard” .
Lower Maquoketa and Galena Group	Sedimentary rock aquitard (consolidated or semi-consolidated rock)	From Prior et al.,2003 page 14: Hydrogeologic units: “Ordovician confining units” Dominant geologic material: “shale, dolomite, chert”, & “dolomite, limestone, chert” . Hydrologic condition: “confining beds, aquitard, local aquifer in northeastern”, & “confining beds, aquitard, local aquifer in northeastern” respectively.
Decorah-Platteville aquitard	Sedimentary rock aquitard (consolidated or semi-consolidated rock)	From Prior et al.,2003 page 14: Hydrogeologic units: “Ordovician confining units” Dominant geologic material: “shale, limestone, sandstone” . Hydrologic condition: “confining beds, aquitard” .
Cambrian-Ordovician Aquifer	St. Peter Sandstone & Jordan Sandstone are (Clastic sedimentary rock aquifer (consolidated or semi-consolidated rock)) and Prairie du Chen Group & St. Lawrence Formation are (Carbonate aquifer)	From Prior et al.,2003 page 14: Hydrogeologic units: “Cambrian-Ordovician aquifer (“Jordan aquifer”)” Dominant geologic material: “sandstone”, “dolomite, sandstone, chert”, “sandstone”, & “dolomite” . Hydrologic condition: “regional aquifer”, regional aquifer”, regional aquifer”, & regional aquifer” respectively.
Cambrian aquitard (Shaly)	Sedimentary rock aquitard (consolidated or semi-consolidated rock)	From Prior et al.,2003 page 14: Hydrogeologic units: “Cambrian confining unit” Dominant geologic material: “shale, siltstone, sandstone” . Hydrologic condition: “confining beds; aquitard” .
Mt. Simon Aquitard	Clastic sedimentary rock aquifer (consolidated or semi-consolidated rock)	From Prior et al.,2003 page 14: Hydrogeologic units: “Dresbach aquifer” Dominant geologic material: “sandstone, shale, dolomite” . Hydrologic condition: “regional aquifer” .
Precambrian Crystalline Basement	Endogenous bedrock	From Prior et al.,2003 page 14: Dominant geologic material: “igneous & metamorphic rocks, sandstone, shale” . Hydrologic condition: “unknown” . “Cambrian and Ordovician rocks form the dominant aquifer system in the study area. They are bounded below by Precambrian basement rocks, which have very low permeability ” (Young, 1992).

Prior, J. C., Boekhoff, J. L., Howes, M. R., Libra, R. D., VanDorpe, P. E. (2003). Iowa's Groundwater Basics: A geological guide to the occurrence, use, and vulnerability of Iowa's aquifers. Accessed June 11, 2022 via https://s-iihr34.iihr.uiowa.edu/publications/uploads/2014-08-24_08-08-21_es-06.pdf

Young, H. L. (1992). *Summary of ground-water hydrology of the Cambrian-Ordovician aquifer system in the northern Midwest, United States* (Vol. 1405). US Department of the Interior, US Geological Survey. Accessed June 11, 2022 via <https://pubs.er.usgs.gov/publication/pp1405B>

Gannon, J. M., McKay, R. M. (2013). *Groundwater Availability Modeling of the Mississippian Aquifer North-Central Iowa* (Vol. 8). University of Iowa. Accessed June 11, 2022 via <https://iro.uiowa.edu/esploro/outputs/report/Groundwater-Availability-Modeling-of-the-Mississippian/9984109923302771>

3.56 Eureka and Eel River and Mad River Plains

Supplementary Fig. 208. Hydrogeologic cross section. 20 equally spaced transparent pink bars overlie the cross section; each shaded bar depicts the vertical offset from the land surface to the top of the uppermost confining unit or endogenous bedrock.

The Eureka and Eel River and Mad River Plains are located in coastal northwest California.

(i) A hydrogeologic cross section presented in Fig. 3 by Johnson (1975) suggests that the aquifer system does not depict a clear confining unit within the aquifer system (see cross section to the left); however, fine-grained interbedded layers lead to confined conditions in the lower portion of the Hookton Formation and Carlotta Formation (see quotes from Johnson (1975) in "(iii)" below).

(ii) We analysed wells within the study area that the USGS has defined as either unconfined or confined. There are no USGS wells within the study area that have been defined as tapping an aquifer that is either unconfined or confined.

(iii) With regards to the presence of confined conditions, a local-scale study by Johnson (1975) states that (quote) *"The two younger units, the Scotia Bluffs Sandstone and Carlotta Formation, consist dominantly of coarse-grained clastic sediments of marginal marine deposition and may be important aquifers. Water found in these two formations in the Eureka area is generally confined by interbeds of silt and clay of low permeability or by the over-lying, fine-grained sediment in the Hookton Formation"*. and, with regards to the Hookton

Formation). Johnson (1975) states (quote) *"The Hookton Formation supplies water to many domestic wells in the Dows Prairie-McKinleyville area and on the hills and terraces of the study area. At lower altitudes and under the river valleys, water in the lower parts of the Hookton is in part confined by the overlying material"*. Because the lower portion of the Hookton Formation is referred to as confined, we estimate the depth to confined conditions to be approximately half the maximum thickness of the Hookton Formation (maximum thickness is ~150 m; half the maximum thickness is ~75 m) as depicted on the cross section by Johnson (1975) (see figure in upper left of this page). We highlight that this approximation is uncertain.

Depth to confined conditions:
70-80 m (see (iii) above)

Reference: Johnson, M. J. (1975). Ground-water conditions in the Eureka Area, Humboldt County, California. U.S. Geological Survey Water-Resources Investigations 78-127. 51 pp. Accessed March 20, 2021 from <https://pubs.usgs.gov/wri/1978/0127/report.pdf>

The table below presents a series of published quotes (see quotation marks denoting text quoted from another publication, which is cited following the quotation marks with the full reference written in full below the table). The leftmost column lists a title of a hydrogeologic formation depicted in the cross section on the previous page. The rightmost column presents a quote from a hydrogeological study (see base of table for citation). The quote has been annotated with colored text to highlight how we categorized each layer (i.e., see categories in the center column in the table). Specifically: (i) blue text highlights portions of a quote that provide insights into the degree of consolidation of the formation, (ii) red text highlights portions of a quote that categorize the formation as an aquifer or an aquitard (i.e., higher versus lower permeability in the context of local hydrogeologic formations), and (iii) green text highlights portions of a quote that provide information about the lithology of the formation.

Supplementary Table 59. Hydrostratigraphy details for Eureka and Eel River and Mad River Plains

Formation name	Category	Quote
Hookton Formation	Unconsolidated aquifer (In some area this aquifer is confined aquifer)	" Poorly consolidated ; yields water to wells in small to moderate amounts from sand and gravel strata. Confined aquifers south of Arcata. " (Johnson, 1978).
Carlotta Formation	Unconsolidated aquifer (In some area this aquifer is confined aquifer)	" Poorly consolidated ; locally yields water to wells in moderate to large amounts from confined sand and gravel" (Johnson, 1978).
Scotia Bluffs Sandstone	Sedimentary rock aquitard (consolidated or semi-consolidated rock)	" Massive, fine-grained sandstone with mudstone member in lower part and fluvial sand and gravel in upper part, predominantly of shallow marine origin. " (Johnson, 1978). " Semi-consolidated to poorly consolidated. Not tapped by water wells, potential yield unknown. " (Johnson, 1978).
Rio Dell, Eel River and Pullen Formation of Ogle	Sedimentary rock aquitard (consolidated or semi-consolidated rock)	" Compact mudstone, claystone, siltstone, and some sandstone, predominantly of marine origin. " (Johnson, 1978). " Semi-consolidated; not tapped by wells, probably poor aquifer. " (Johnson, 1978).

Johnson, M. J. (1978). *Ground-water conditions in the Eureka area, Humboldt County, California, 1975* (Vol. 78). US Geological Survey, Water Resources Division.

3.57 Garber-Wellington Aquifer

Supplementary Fig. 209. Hydrogeologic cross section. 20 equally spaced transparent pink bars overlie the cross section; each shaded bar depicts the vertical offset from the land surface to the top of the uppermost confining unit or endogenous bedrock.

Supplementary Fig. 210. Vertical variations in the prevalence of wells that have been defined as tapping an unconfined or a confined aquifer by the USGS. The smaller squares represent 10 m depth intervals from the land surface to 100 m; the larger squares represent 20 m intervals from 100 m to 300 m below the land surface.

The Garber-Wellington aquifer system is located in central Oklahoma.

(i) A hydrogeologic cross section presented in Fig. 5 by Mashburn et al. (2014) does not depict a clear confining unit in the central portion of the study area (but see quote in (iii) below).

(ii) We analysed wells within the study area that the USGS has defined as either unconfined or confined. Most (>80%) wells at depths of 80-90 m and at depths exceeding 80 m are defined as tapping a confined aquifer.

(iii) Mashburn et al. (2014) state that (following text quoted directly): "...even though the Central Oklahoma aquifer extends to land surface with a potentiometric surface below the top of the Central Oklahoma aquifer, the groundwater system acts as a confined system due to laterally extensive interbedded mudstone and large contrasts in vertical hydraulic conductivity."

Depth to confined conditions: 80-90 m (see (ii) above)

Reference: Mashburn, S.L., Ryter, D.W., Neel, C.R., Smith, S.J., Correll, J.S., (2014). Hydrogeology and simulation of ground-water flow in the Central Oklahoma (Garber-Wellington) Aquifer, Oklahoma, 1987 to 2009, and simulation of available water in storage, 2010–2059. US Geological Survey Scientific Investigations Report 2013–5219, 92 pp. Accessed May 14, 2021 from https://pubs.usgs.gov/sir/2013/5219/pdf/sir20135219_v2.0.pdf

The table below presents a series of published quotes (see quotation marks denoting text quoted from another publication, which is cited following the quotation marks with the full reference written in full below the table). The leftmost column lists a title of a hydrogeologic formation depicted in the cross section on the previous page. The rightmost column presents a quote from a hydrogeological study (see base of table for citation). The quote has been annotated with colored text to highlight how we categorized each layer (i.e., see categories in the center column in the table). Specifically: (i) blue text highlights portions of a quote that provide insights into the degree of consolidation of the formation, (ii) red text highlights portions of a quote that categorize the formation as an aquifer or an aquitard (i.e., higher versus lower permeability in the context of local hydrogeologic formations), and (iii) green text highlights portions of a quote that provide information about the lithology of the formation.

Supplementary Table 60. Hydrostratigraphy details for the Garber-Wellington aquifer

Formation name	Category	Quote
Hennessey Group (confining unit) (some small-yield wells exist)	Sedimentary rock aquitard (consolidated or semi-consolidated rock)	"The Permian-age Hennessey Group overlies the western part of the Central Oklahoma aquifer and consists of interbedded red shale, clay, and some siltstone or fine-grained sandstone (Parkhurst and others, 1996)." (Mashburn, et al., 2014). "the Hennessey Group outcrops in the western one-third of the aquifer area and acts as a confining layer because of its small transmissivity." (Mashburn, et al., 2014).
Central Oklahoma aquifer (referred locally as the Garber-Wellington aquifer. Quaternary deposit is often unconsolidated)	Clastic sedimentary rock aquifer (consolidated or semi-consolidated rock) Quaternary deposit of the Central Oklahoma aquifer is unconsolidated.	"The Garber Sandstone and Wellington Formation consist of cross-bedded, fine-grained sandstone with interbedded shale or mudstone (Bingham and Moore, 1975; Parkhurst and others, 1996; Breit, 1998)." (Mashburn, et al., 2014). "The Central Oklahoma aquifer consists of Quaternary-age alluvium and terrace deposits and Permian-age geologic units (figs. 4 and 5 and table 2) (Christenson and others, 1990). Groundwater flows between these geologic units and both the Quaternary-age and Permian-age units are used as a source of groundwater." (Mashburn, et al., 2014).
Chase, Council Grove and Admire Group	Clastic sedimentary rock aquifer (consolidated or semi-consolidated rock)	"The Chase, Council Grove, and Admire Groups are composed of cross-bedded, fine-grained sandstone, shale, and thin limestone (Bingham and Moore, 1975)." (Mashburn, et al., 2014). "Groundwater-flow in the Central Oklahoma aquifer is slowest in the confined part of the Garber Sandstone and Wellington Formation and in the less transmissive parts of the unconfined flow system, which includes part of the Chase, Council Grove, and Admire Groups." (Mashburn, et al., 2014).
Pennsylvanian geologic units, undifferentiated (includes the Vanoss Formation)	Sedimentary rock aquitard (consolidated or semi-consolidated rock)	"the Vanoss Formation is considered to be a hydrologic boundary of little to no flow." (Mashburn, et al., 2014). "The Vanoss Formation consists of red-brown to grey shale and intermittent thin limestone and sandstone beds that range in total thickness from 250 to 490 feet (Bingham and Moore, 1975). This formation acts as a lower confining unit to the Central Oklahoma aquifer, limiting vertical groundwater movement." (Mashburn, et al., 2014).

Mashburn, S.L., Ryter, D.W., Neel, C.R., Smith, S.J., Correll, J.S., (2014). Hydrogeology and simulation of ground-water flow in the Central Oklahoma (Garber-Wellington) Aquifer, Oklahoma, 1987 to 2009, and simulation of available water in storage, 2010–2059. US Geological Survey Scientific Investigations Report 2013–5219, 92 pp. Accessed June 5, 2022 from <https://pubs.er.usgs.gov/publication/sir20135219>

3.58 Honey Lake Valley

Supplementary Fig. 211. Hydrogeologic cross section. 20 equally spaced transparent pink bars overlie the cross section; each shaded bar depicts the vertical offset from the land surface to the top of the uppermost confining unit or endogenous bedrock.

Honey Lake Valley is located in northeastern California.

(i) A hydrogeologic cross section presented in Fig. 2 by Mayo et al. (2010) suggests that the unconfined portion of Honey Lake Valley extends to a typical depth of greater than 683 m meters below land surface (median of pink bars shown in cross section to the left).

(ii) We could not identify a sufficient number of USGS wells that have been defined as tapping either unconfined or confined conditions.

(iii) In a local-scale study, Mayo et al. (2010) state that (quote) "Three groundwater systems have been identified in Honey Lake basin: (1) shallow unconfined and semiconfined (<200 m below ground surface (bgs)), (2) deep confined (>200 m bgs), and (3) geothermal". This quote is consistent with the approximate depth to the top of the Pliocene Lake Deposits presented in the cross section (which exist at a median depth of 162.5 meters below land surface).

Depth to confined conditions: 200-220 m based on (iii)

Reference: Mayo, A. L., Henderson, R. M., Tingey, D., Webber, W. (2010). Chemical evolution of shallow playa groundwater in response to post-pluvial isostatic rebound, Honey Lake Basin, California-Nevada, USA. Hydrogeology Journal, 18, 725-747.

The table below presents a series of published quotes (see quotation marks denoting text quoted from another publication, which is cited following the quotation marks with the full reference written in full below the table). The leftmost column lists a title of a hydrogeologic formation depicted in the cross section on the previous page. The rightmost column presents a quote from a hydrogeological study (see base of table for citation). The quote has been annotated with colored text to highlight how we categorized each layer (i.e., see categories in the center column in the table). Specifically: (i) blue text highlights portions of a quote that provide insights into the degree of consolidation of the formation, (ii) red text highlights portions of a quote that categorize the formation as an aquifer or an aquitard (i.e., higher versus lower permeability in the context of local hydrogeologic formations), and (iii) green text highlights portions of a quote that provide information about the lithology of the formation.

Supplementary Table 61. Hydrostratigraphy details for Honey Lake Valley

Formation name	Category	Quote
Recent Lake and Lake Lahontan Deposits	Unconsolidated aquifer (unconfined aquifer, and semi-confined aquifer)	"Honey Lake basin is filled with up to 1,700 m of Pliocene and younger volcanic tuff and ash, and terrestrial, fluvial, and lacustrine sand, silt and clay (Fig. 2; Handman et al. 1990; California Department of Water Resources 1963, 2003)" (Mayo et al., 2010). "The uppermost 300 m of basin sediments are pluvial (Pleistocene Lake Lahontan), fluvial and deltaic deposits (Bonham 1969; Grose et al. 1990; Handman et al. 1990)." "The pluvial lake and basin margin sediments support the shallow groundwater system described in this investigation." (Mayo et al., 2010).
Pliocene Lake Deposits	Clastic sedimentary rock aquifer (consolidated or semi-consolidated rock)	"Honey Lake basin is filled with up to 1,700 m of Pliocene and younger volcanic tuff and ash, and terrestrial, fluvial, and lacustrine sand, silt and clay (Fig. 2; Handman et al. 1990; California Department of Water Resources 1963, 2003)" (Mayo et al., 2010). "The uppermost 300 m of basin sediments are pluvial (Pleistocene Lake Lahontan), fluvial and deltaic deposits (Bonham 1969; Grose et al. 1990; Handman et al. 1990)." (Mayo et al., 2010). "The pluvial lake and basin margin sediments support the shallow groundwater system described in this investigation." (Mayo et al., 2010).
Alluvial Fan Deposits	Unconsolidated aquifer (unconfined aquifer, and semi-confined aquifer)	"Elsewhere poorly sorted, highly permeable alluvial fan deposits , derived from granodiorite and volcanic terrains, flank the basin and interfinger with Lake Lahontan and younger basin deposits (California Department of Water Resources 2003; Handman et al. 1990)." (Mayo et al., 2010)
Volcanic Rocks	Volcanic aquifer (local permeable region exists)	"Elsewhere volcanic rocks consist of locally transmissive lava flows (rhyolite, andesite, and basalt), tuff, flow breccia, and volcanic breccia and conglomerate. " (Mayo et al., 2010).
Granitic Bedrock	Endogenous bedrock (thermal water and local permeable region can exist, discharge to spring)	"The granitic terrains are locally overlain by Pliocene–Miocene age volcanic rocks (rhyolite, dacite, andesite, and tuff) that contain phenocrysts of plagioclase (An50–70), pyroxene, hornblende, and olivine in varying proportions (Grose 1984, 1993; Grose et al. 1990, 1993)." (Mayo et al., 2010). " Granitic Bedrock ". (Mayo et al., 2010).

Mayo, A. L., Henderson, R. M., Tingey, D., & Webber, W. (2010). Chemical evolution of shallow playa groundwater in response to post-pluvial isostatic rebound, Honey Lake Basin, California–Nevada, USA. *Hydrogeology Journal*, **18**, 725-747.

3.59 Long Island

Supplementary Fig. 212. Hydrogeologic cross section. 20 equally spaced transparent pink bars overlie the cross section; each shaded bar depicts the vertical offset from the land surface to the top of the uppermost confining unit or endogenous bedrock.

Supplementary Fig. 213. Vertical variations in the prevalence of wells that have been defined as tapping an unconfined or a confined aquifer by the USGS. The smaller squares represent 10 m depth intervals from the land surface to 100 m; the larger squares represent 20 m intervals from 100 m to 300 m below the land surface.

Long Island is located in eastern New York.

(i) A hydrogeologic cross section presented in Plate 1 by Smolensky et al. (1990) suggests that the aquifer system has two primary confining units, the Gardiners Clay and the Raritan Confining Unit.

(ii) We analysed wells within the study area that the USGS has defined as either unconfined or confined. Most (>80%) wells at depths of 120-140 m and at depths exceeding 120 m are defined as tapping a confined aquifer.

Depth to confined conditions: 120-140 m (see (ii) above)

Reference: Smolensky, D.A., Buxton, H.T., Shernoff, P.K. (1990). Hydrologic framework of Long Island, New York. U.S. Geological Survey Hydrologic Atlas 709, 3 plates. Accessed June 15, 2022 from <https://pubs.usgs.gov/ha/709/plate-1.pdf>

The table below presents a series of published quotes (see quotation marks denoting text quoted from another publication, which is cited following the quotation marks with the full reference written in full below the table). The leftmost column lists a title of a hydrogeologic formation depicted in the cross section on the previous page. The rightmost column presents a quote from a hydrogeological study (see base of table for citation). The quote has been annotated with colored text to highlight how we categorized each layer (i.e., see categories in the center column in the table). Specifically: (i) blue text highlights portions of a quote that provide insights into the degree of consolidation of the formation, (ii) red text highlights portions of a quote that categorize the formation as an aquifer or an aquitard (i.e., higher versus lower permeability in the context of local hydrogeologic formations), and (iii) green text highlights portions of a quote that provide information about the lithology of the formation.

Supplementary Table 62. Hydrostratigraphy details for the Long Island

Formation name	Category	Quote
UG-Upper glacial aquifer	Unconsolidated aquifer	Smolensky et al., 1990 Plate 1, table 1 " Till composed of clay, sand, gravel, and boulders, forms Harbor Hill and Ronkonkoma terminal moraines. Outwash deposits consist of quartzose sand, fine to very coarse, and gravel, pebble to boulder sized. Also contains lacustrine, marine, and reworked deposits. Local units are Port Washington aquifer and confining unit, "20-foot clay, and clay at Smithtown.". Smolensky et al., 1990 Plate 1, table 1 " Till is poorly permeable. Outwash deposits are moderately to highly permeable. Glaciolacustrine and marine clay deposits are mostly poorly permeable but locally have thin, moderately permeable layers of sand and gravel. Average horizontal hydraulic conductivity is approximately 270 ft/d; conductivity of morainal material is approximately 50 percent of outwash deposits, anisotropy is approximately 10:1. "
Gc – Gardiners Clay	Sedimentary rock aquitard (consolidated or semi-consolidated rock)	Smolensky et al., 1990 Plate 1, table 1 " Clay, silt, and few layers of sand. Colors are grayish green and brown. Contains marine shells and glauconite. ". Smolensky et al., 1990 Plate 1, table 1 " Poorly permeable; constitutes a confining layer for underlying aquifer. Some sand lenses may be permeable Average vertical hydraulic conductivity is approximately 0.001 ft/d. "
Mg – Monmouth greensand	Sedimentary rock aquitard (consolidated or semi-consolidated rock)	Smolensky et al., 1990 Plate 1, table 1 " Interbedded marine deposits of clay, silt, and sand. Dark-greenish gray, Greenish-black, greenish, dark-gray. And black, containing much glauconite. ". Smolensky et al., 1990 Plate 1, table 1 " Poorly permeable; primarily a confining unit for underlying Magothy aquifer. Average vertical hydraulic conductivity is approximately 0.001 ft/d. "
M – Magothy aquifer	Sedimentary rock aquifer (consolidated or semi-consolidated rock)	Smolensky et al., 1990 Plate 1, table 1 " Sand, fine to medium clayey in part: interbedded with lenses and layers of coarse sand and sandy and solid clay. Gravel is common in basal zone, Sand and gravel are quartzose. Lignite, pyrite, and iron oxide concretions are common. Colors are gray, white, red brown, and yellow.". Smolensky et al., 1990 Plate 1, table 1 " Most layers are poorly to moderately permeable; some are highly permeable locally. Water is unconfined in uppermost parts, elsewhere is confined. Constitutes principal aquifer for

Formation name	Category	Quote
		public supply. Average horizontal hydraulic conductivity is 50 ft/d; anistrophy is approximately 100:1."
Rc – Raritan confining unit	Sedimentary rock aquitard (consolidated or semi-consolidated rock)	Smolensky et al., 1990 Plate 1, table 1 " Clay, solid and silty; few lenses and layers of sand. Lignite and pyrite are common. Colors are gray, red, and white, commonly variegated." Smolensky et al., 1990 Plate 1, table 1 " Poorly to very poorly permeable; constitutes confining layer for underlying Lloyd aquifer. Average vertical hydraulic conductivity is approximately 0.001 ft/d."
L – Lloyd aquifer	Sedimentary rock aquifer (consolidated or semi-consolidated rock)	Smolensky et al., 1990 Plate 1, table 1 " Sand, fine to coarse, and gravel, commonly with clayey matrix; some lenses and layers of solid and silty clay; locally contains thin lignite layers. Sand and most of gravel are quartzose. Colors are yellow, gray, and white; clay is red locally." Smolensky et al., 1990 Plate 1, table 1 " Poorly to moderately permeable. Water is confined by overlying Raritan clay. Average horizontal hydraulic conductivity is 40ft/d; anisotropy is approximately 10:1."
Br – Bedrock	Endogenous bedrock	Smolensky et al., 1990 Plate 1, table 1 " Crystalline metamorphic and igneous rocks: muscovite-biotite schist, gneiss, and granite. A soft, clayey zone of weathered bedrock locally is more than 70 ft thick." Smolensky et al., 1990 Plate 1, table 1 " Poorly permeable to virtually impermeable; constitutes lower boundary of ground-water reservoir. Some hard fresh water is contained in joints and fractures but is impractical to develop at most places."

Smolensky, D.A., Buxton, H.T., Shernoff, P.K. (1990). Hydrologic framework of Long Island, New York, U.S. Geological Survey Hydrologic Atlas 709, 3 plates. Accessed June 15, 2022 from <https://pubs.usgs.gov/ha/709/plate-1.pdf>

Olcott, P.G. (1995). Ground Water Atlas of the United States: Segment 12, Connecticut, Maine, Massachusetts, New Hampshire, New York, Rhode Island, Vermont. U.S. Geological Survey Hydrologic Atlas 730-M, 30 pp. Accessed April 14, 2021 from <https://pubs.usgs.gov/ha/730m/report.pdf>

3.60 Los Angeles Basin

Supplementary Fig. 214. Hydrogeologic cross section. 20 equally spaced transparent pink bars overlie the cross section; each shaded bar depicts the vertical offset from the land surface to the top of the uppermost confining unit or endogenous bedrock.

Supplementary Fig. 215. Vertical variations in the prevalence of wells that have been defined as tapping an unconfined or a confined aquifer by the USGS. The smaller squares represent 10 m depth intervals from the land surface to 100 m; the larger squares represent 20 m intervals from 100 m to 300 m below the land surface.

The Los Angeles Basin is located in coastal southern California.

(i) A hydrogeologic cross section presented in Fig. 4 by Reichard et al. (2003) does not depict a continuous confining unit in the study area.

(ii) We analysed wells within the study area that the USGS has defined as either unconfined or confined. We could not identify a depth where most (>80%) wells with the depth range and at deeper depths are defined as tapping a confined aquifer. Three of the four deepest wells in our dataset (depths of 396 m, 399 m, 445 m, and 472 m) are classified as unconfined.

Depth to confined conditions: >472 m (see (ii) above)

Reference: Reichard, E. G., Land, M., Crawford, S.M., Johnson, T.D., Everett, R.R., Kulshan, T.V., Ponti, D.J., Halford, K.L., Johnson, T.A., Paybins, K.S., Nishikawa, T. (2003). *Geohydrology, Geochemistry, and Ground-Water Simulation Optimization of the Central and West Coast Basins, Los Angeles County, California.* USGS Numbered Series, 2003-4065. Accessed November 28, 2021 from [https://pubs.usgs.gov/wri/wrir034065.pdf](https://pubs.usgs.gov/wri/wrir034065/wrir034065.pdf)

The table below presents a series of published quotes (see quotation marks denoting text quoted from another publication, which is cited following the quotation marks with the full reference written in full below the table). The leftmost column lists a title of a hydrogeologic formation depicted in the cross section on the previous page. The rightmost column presents a quote from a hydrogeological study (see base of table for citation). The quote has been annotated with colored text to highlight how we categorized each layer (i.e., see categories in the center column in the table). Specifically: (i) blue text highlights portions of a quote that provide insights into the degree of consolidation of the formation, (ii) red text highlights portions of a quote that categorize the formation as an aquifer or an aquitard (i.e., higher versus lower permeability in the context of local hydrogeologic formations), and (iii) green text highlights portions of a quote that provide information about the lithology of the formation.

Supplementary Table 63. Hydrostratigraphy details for Los Angeles Basin

Formation name	Category	Quote
Recent	Unconsolidated aquifer	"The geohydrologic units that compose the Holocene (Recent) age deposits of the Recent aquifer system include the semiperched aquifer, the Bellflower aquiclude, the Gaspur aquifer, and the Ballona aquifer (California Department of Water Resources, 1961). Although these geohydrologic units are referred to in this report as consisting of Holocene-age deposits , some of these units consist of deposits of Pleistocene age. The semiperched aquifer is a relatively thin layer of coarse sand and gravel near the land surface; it consists of alluvial sediments and, in parts of the West Coast Basin, marine deposits that may include the late Pleistocene Palos Verdes Sand. " (Reichard et al., 2003).
Lakewood	Unconsolidated aquifer	"Generally, the Lakewood aquifer system is a heterogeneous unit dominated by sandy silts and silty sands interbedded with sands that become coarser and thicker near the base of the aquifer system. " (Reichard et al., 2003).
Upper San Pedro	Clastic sedimentary rock aquifer (consolidated or semi-consolidated rock)	"The Upper San Pedro aquifer system incorporates the Hollydale, Jefferson, Lynwood, and Silverado aquifers (fig. 3)." (Reichard et al., 2003). "Overall, the aquifer system appears to be of mixed origin, with nonmarine deposits consisting of sand and gravel that are interbedded with silt and clay, and marine deposits characterized by blue-gray sand, gravel, silt, and clay, along with shells and wood fragments. " (Reichard et al., 2003).
Lower San Pedro	Clastic sedimentary rock aquifer (consolidated or semi-consolidated rock)	"The Lower San Pedro aquifer system includes the Sunnyside aquifer (also referred to as the Lower San Pedro aquifer) . The upper part of this system tends to be characterized by alternating fine-grained and coarse-grained zones . The fine-grained zones tend to pinch out or disappear near the forebay margins, such as at USGS Pico Rivera-1 (2S/11W-18C4-7) and 1S/13W-34F (fig. 4A,B). The coarsest part of the aquifer system generally is at the base and is as much as 100 ft thick. " (Reichard et al., 2003).

Formation name	Category	Quote
Pico unit	Clastic sedimentary rock aquifer (consolidated or semi-consolidated rock)	"The Pico aquifer, which underlies the entire area from Bolsa Chica Mesa to Newport Mesa, ranges in thickness from about 1,200 feet at Huntington Beach Mesa to less than 1 foot at Newport Mesa. The permeable zones in the aquifer are generally of fine to medium sand with permeability ranging from 200 to 300 gpd / ft 2 (gallons per day per square foot)." (Moreland and Singer, 1969).

Reichard, E. G., Land, M., Crawford, S.M., Johnson, T.D., Everett, R.R., Kulshan, T.V., Ponti, D.J., Halford, K.L., Johnson, T.A., Paybins, K.S., Nishikawa, T. (2003). Geohydrology, Geochemistry, and Ground-Water Simulation Optimization of the Central and West Coast Basins, Los Angeles County, California. USGS Numbered Series, 2003-4065. Accessed June 10, 2022 via <https://pubs.usgs.gov/wri/wrir034065/wrir034065.pdf>

Moreland, J. A., Singer, J. A. (1969). A study of deep aquifers underlying coastal Orange County, California. US Geological Survey, Water Resources Division. Accessed June 10, 2022 via <https://pubs.er.usgs.gov/publication/70047681>

3.61 Michigan Basin

Supplementary Fig. 216. Hydrogeologic cross section. 20 equally spaced transparent pink bars overlie the cross section; each shaded bar depicts the vertical offset from the land surface to the top of the uppermost confining unit or endogenous bedrock.

Supplementary Fig. 217. Vertical variations in the prevalence of wells that have been defined as tapping an unconfined or a confined aquifer by the USGS. The smaller squares represent 10 m depth intervals from the land surface to 100 m; the larger squares represent 20 m intervals from 100 m to 300 m below the land surface.

The Michigan Basin is located in central Michigan.

(i) A hydrogeologic cross section presented in Fig. 7 by Westjohn et al. (1998) depicts a series of stacked confining units underlying relatively thick (>100 m) glacial deposits.

(ii) We analysed wells within the study area that the USGS has defined as either unconfined or confined. Most (>80%) wells at depths of 40-50 m and at depths exceeding 40 m are defined as tapping a confined aquifer.

Depth to confined conditions: 40-50 m (see (ii) above)

Reference: Westjohn, D.B., Weaver, T.L. (1998). Hydrogeologic framework of the Michigan Basin regional aquifer system. US Geological Survey Professional Paper 1418, 55 pp. Accessed November 29, 2021 from <https://pubs.usgs.gov/pp/1418/report.pdf>

The table below presents a series of published quotes (see quotation marks denoting text quoted from another publication, which is cited following the quotation marks with the full reference written in full below the table). The leftmost column lists a title of a hydrogeologic formation depicted in the cross section on the previous page. The rightmost column presents a quote from a hydrogeological study (see base of table for citation). The quote has been annotated with colored text to highlight how we categorized each layer (i.e., see categories in the center column in the table). Specifically: (i) blue text highlights portions of a quote that provide insights into the degree of consolidation of the formation, (ii) red text highlights portions of a quote that categorize the formation as an aquifer or an aquitard (i.e., higher versus lower permeability in the context of local hydrogeologic formations), and (iii) green text highlights portions of a quote that provide information about the lithology of the formation.

Supplementary Table 64. Hydrostratigraphy details for the Michigan Basin

Formation name	Category	Quote
Drift – Glacial deposits (undifferentiated) (fine grained glacial lacustrine and till are assumed to function as confining units.)	Unconsolidated aquifer	"Glacial deposits in the study area can be separated into three general provinces (fig. 10): (1) glacial deposits in the southern part are primarily recessional moraines and outwash deposits that formed at the front of retreating ice lobes; (2) surficial deposits in the Saginaw Lowlands are primarily basal-lodgment tills and fine grained lacustrine sediments that were deposited in former proglacial lakes; and (3) glacial deposits of the northern half of the study area are primarily glaciofluvial deposits and some coarse-textured till (Farrand and Bell, 1982)." (Westjohn & Weaver, 1998). " Glaciofluvial deposits are the largest source of fresh ground water in Michigan and in the RASA study area." (Westjohn & Weaver, 1998).
Red Beds	Sedimentary rock aquitard (consolidated or semi-consolidated rock)	"Shaffer (1969) indicates that predominant lithologies of the "red beds" sequence are red clay, mudstone, siltstone, and sandstone; as well as gray-green shale, mudstone, and gypsum. " (Westjohn & Weaver, 1998). "On the basis of data from geophysical logs, permeable sandstone within Jurassic "red beds" is not volumetrically important and most of the sequence is probably of low permeability. " (Westjohn & Weaver, 1998).
Grand River – Saginaw Formation	Sedimentary rock aquifer (consolidated or semi-consolidated rock)	"The Grand River Formation is reported to consist predominantly of sandstone, although Kelly (1936, p. 209) indicates that a conglomerate bed at the type locality separates the Grand River Formation from the underlying Saginaw Formation." (Westjohn & Weaver, 1998). "Hydrogeologic units that include all or parts of two stratigraphic units are the Saginaw aquifer (sandstones of the Grand River Formation and the Saginaw Formation), the Parma-Bayport aquifer (sandstones and permeable carbonates of the Parma Sandstone and the Bayport Limestone), and the Marshall aquifer (composite of stratigraphically continuous, permeable sandstones of the Michigan Formation and the Marshall Sandstone)." (Westjohn & Weaver, 1998).
Saginaw Formation (principally shale) (underlying Saginaw shale	Sedimentary rock aquitard (consolidated or semi-consolidated rock)	"The Saginaw confining unit separates the Saginaw aquifer from the underlying Parma-Bayport aquifer in most of the study area (fig. 5). This confining unit consists mostly of shale; the rest of the sequence consists of thin beds of sandstone, siltstone, coal, and limestone. " (Westjohn & Weaver, 1998).

Formation name	Category	Quote
dominated unit is confining layer)		
Parma Sandstone	Sedimentary rock aquifer (consolidated or semi-consolidated rock)	"The Parma Sandstone , which consists of medium- to coarse-grained sandstone , is typically less than 100 ft thick (Cohee and others, 1951)." (Westjohn & Weaver, 1998). "Geophysical logs show that these units consist mostly of permeable sandstones and carbonates and that the formations are hydraulically connected throughout the area of the regional aquifer system. For characterization of hydrogeologic setting and computer simulation of groundwater flow, these units are combined as the Parma-Bayport aquifer ." (Westjohn & Weaver, 1998).
Bayport Limestone	Carbonate aquifer	"The Bayport Limestone of Late Mississippian age consists of sparsely fossiliferous to highly fossiliferous limestone, dolostone, sandy limestone, cherty limestone, and sandstone (Bacon, 1971; Ciner, 1988; Lasemi, 1975; Tyler, 1980)." (Westjohn & Weaver, 1998). "Hydrogeologic units that include all or parts of two stratigraphic units are the Saginaw aquifer (sandstones of the Grand River Formation and the Saginaw Formation), the Parma-Bayport aquifer (sandstones and permeable carbonates of the Parma Sandstone and the Bayport Limestone), and the Marshall aquifer (composite of stratigraphically continuous, permeable sandstones of the Michigan Formation and the Marshall Sandstone)." (Westjohn & Weaver, 1998).
Michigan Formation	Sedimentary rock aquitard (consolidated or semi-consolidated rock)	"The Michigan confining unit is composed of all low permeability lithologies of the Michigan Formation, and does not include stratigraphically continuous sandstones at or near the base of the formation. This confining unit separates the Parma-Bayport aquifer from the Marshall aquifer (fig. 5). The Michigan confining unit consists of shale, carbonate, evaporite, and thin, laterally discontinuous siltstone and sandstone lenses ." (Westjohn & Weaver, 1998).
Marshall Sandstone (Napoleon member, lower Marshall unit, and confining unit lithologies) (sand matrix can limit groundwater flow)	Sedimentary rock aquifer (consolidated or semi-consolidated rock)	" Sandstone forms only part of the formation. Limestone, dolomite, siltstone, and shale are interbedded with sandstones of the Marshall sedimentary sequence in different parts of the basin." (Westjohn and Weaver, 1998). "The Marshall aquifer consists of one or more blanket type sandstones of Mississippian age that RASA investigators assume are hydraulically connected at the scale of the regional aquifer system." (Westjohn & Weaver, 1998).
Coldwater Shale	Sedimentary rock aquitard (consolidated or semi-consolidated rock)	"The Coldwater confining unit is the base of the regional aquifer system. This basal confining unit consists mostly of shale, Siltstone, sandstone, limestone, and dolomite are part of this hydrogeologic unit in some areas of the basin." (Westjohn & Weaver, 1998).

Westjohn, D.B., Weaver, T.L. (1998). Hydrogeologic framework of the Michigan Basin regional aquifer system. US Geological Survey Professional Paper 1418, 55 pp. Accessed November 29, 2021 from <https://pubs.usgs.gov/pp/1418/report.pdf>

3.62 Mojave Basin

Supplementary Fig. 218. Hydrogeologic cross section. 20 equally spaced transparent pink bars overlie the cross section; each shaded bar depicts the vertical offset from the land surface to the top of the uppermost confining unit or endogenous bedrock.

Supplementary Fig. 219. Vertical variations in the prevalence of wells that have been defined as tapping an unconfined or a confined aquifer by the USGS. The smaller squares represent 10 m depth intervals from the land surface to 100 m; the larger squares represent 20 m intervals from 100 m to 300 m below the land surface.

The Mojave Basin is located in southern California.

(i) A hydrogeologic cross section presented in Fig. 2a by Kulongoski et al. (2003) does not depict a clear confining unit within the aquifer system. The median depth to basement

igneous and metamorphic rocks as depicted in the cross section is >512 m.

(ii) We could not identify a depth where most (>80%) wells with the depth range and at deeper depths are defined as tapping a confined aquifer. The two deepest wells in our dataset (depths of 228 m and 314 m) are classified as unconfined.

(iii) Kulongoski et al. (2003) highlight that unconfined conditions prevail in parts of the aquifer system; they state (following text quoted directly): "These deposits (QTa) consist of unconsolidated to moderately consolidated gravel, sand, silt, and clay deposited in the Pleistocene and late Pliocene, and overlie a crystalline complex of igneous and metamorphic rocks (pTb). (California Dept. Of Water Resources, 1967). Waters from this mostly unconfined aquifer..."

Depth to confined conditions: >512 m (see (i) and (ii) above)

Reference: Kulongoski, J.T., Hilton, D.R., Izbicki, J.A. (2003). Helium isotope studies in the Mojave Desert, California: implications for groundwater chronology and regional seismicity. *Chemical Geology*, 202, 95-113.

The table below presents a series of published quotes (see quotation marks denoting text quoted from another publication, which is cited following the quotation marks with the full reference written in full below the table). The leftmost column lists a title of a hydrogeologic formation depicted in the cross section on the previous page. The rightmost column presents a quote from a hydrogeological study (see base of table for citation). The quote has been annotated with colored text to highlight how we categorized each layer (i.e., see categories in the center column in the table). Specifically: (i) blue text highlights portions of a quote that provide insights into the degree of consolidation of the formation, (ii) red text highlights portions of a quote that categorize the formation as an aquifer or an aquitard (i.e., higher versus lower permeability in the context of local hydrogeologic formations), and (iii) green text highlights portions of a quote that provide information about the lithology of the formation.

Supplementary Table 65. Hydrostratigraphy details for the Mojave Basin

Formation name	Category	Quote
Undifferentiated alluvium	Unconsolidated aquifer	"The undifferentiated alluvium (QTa), which forms the regional aquifer , is more than 1000 m thick at some locations (Subsurface Surveys, 1990), and consists of alluvial and basin-fill deposits ." Kulongoski et al., (2003)
Igneous and metamorphic basement complex	Endogenous bedrock	"The pre-Tertiary basement complex typically has low porosity and permeability , yielding only small quantities of water to wells ; however, where the basement complex is intensely fractured, as along major faults, the bedrock is more permeable." Stamos et al., (2001)

Kulongoski, J.T., Hilton, D.R., Izbicki, J.A. (2003). Helium isotope studies in the Mojave Desert, California: implications for groundwater chronology and regional seismicity. Chemical Geology, 202(1-2), 95-113.

Stamos, C.L., Martin, P., Nishikawa, T., Cox, B.F. (2001). Simulation of ground-water flow in the Mojave River Basin, California. US Geological Survey Water-Resources Investigations Report, 01-4002.

3.63 Northern Green River Basin

Supplementary Fig. 220. Hydrogeologic cross section. 20 equally spaced transparent pink bars overlie the cross section; each shaded bar depicts the vertical offset from the land surface to the top of the uppermost confining unit or endogenous bedrock.

Supplementary Fig. 221. Vertical variations in the prevalence of wells that have been defined as tapping an unconfined or a confined aquifer by the USGS. The smaller squares represent 10 m depth intervals from the land surface to 100 m; the larger squares represent 20 m intervals from 100 m to 300 m below the land surface.

The Northern Green River Basin is located in western Wyoming.

(i) Fig. 7 by Bartos et al. (2015) suggests that the top of the uppermost confining unit is typically 193 meters below land surface (i.e., the median length of the pink transparent bars in cross section; the lower-upper quartile range is 137-225 meters).

(ii) The available USGS well data are insufficient to evaluate the depths at which the aquifer system transitions from unconfined to confined conditions. Wells with depths of 335 m and 416 m are classified as confined, where the shallowest wells (all with depths of less than 26 m) are classified as unconfined.

(iii) Bartos et al. (2015) state, with regards to the presence of confining units, that (quote) "Confining units are the Wilkins Peak and Tipton confining units; where impermeable, the Laney Member of the Green River Formation also is a confining unit." As the Laney Formation overlies the Wilkins Formation, it is possible that confined conditions prevail at shallower depths than the top of the Wilkins Formation. Our estimated depth to confined conditions may overestimate actual depths to confined conditions in some areas.

Depth to confined conditions: 180-200 meters based on (i)

Reference: Bartos, T.T., Hallberg, L.L., Eddy-Miller, C.A., (2015). Hydrogeology, groundwater levels, and generalized potentiometric-surface map of the Green River Basin Lower Tertiary aquifer system, 2010–14, in the northern Green River structural basin, Wyoming. US Geological Survey Scientific Investigations Report 2015–5090, 33 pp. Accessed on May 13, 2021 from <https://pubs.usgs.gov/sir/2015/5090/sir20155090.pdf>

The table below presents a series of published quotes (see quotation marks denoting text quoted from another publication, which is cited following the quotation marks with the full reference written in full below the table). The leftmost column lists a title of a hydrogeologic formation depicted in the cross section on the previous page. The rightmost column presents a quote from a hydrogeological study (see base of table for citation). The quote has been annotated with colored text to highlight how we categorized each layer (i.e., see categories in the center column in the table). Specifically: (i) blue text highlights portions of a quote that provide insights into the degree of consolidation of the formation, (ii) red text highlights portions of a quote that categorize the formation as an aquifer or an aquitard (i.e., higher versus lower permeability in the context of local hydrogeologic formations), and (iii) green text highlights portions of a quote that provide information about the lithology of the formation.

Supplementary Table 66. Hydrostratigraphy details for Northern Green River Basin

Formation name	Category	Quote
Bridge aquifer	Unconsolidated aquifer	" Aquifers , from top to bottom, are the Bridger, Laney, Farson Sandstone-Alkali Creek, and Wasatch-Fort Union aquifers (fig. 6)." (Bartos, et al., 2015). "Complex intertonguing fluvial and lacustrine sediments . The Wasatch Formation, the Fowkes/Bridger Formations in the southwest Overthrust and Green River Basins near outcrop, as well as Ft Union in Great Divide, Washakie, and Little Snake Basins are major aquifers." States West Water Resources Corporation (2001)
Laney aquifer	Unconsolidated aquifer (it could be confining unit where impermeable. Groundwater in the sandstone aquifers is under unconfined (water table) and confined (artesian) conditions.)	" Aquifers , from top to bottom, are the Bridger, Laney, Farson Sandstone-Alkali Creek, and Wasatch-Fort Union aquifers (fig. 6)." (Bartos, et al., 2015). "The Laney Member of the Green River Formation consists of interbedded layers and lenses of oil shale; marlstone; tuffaceous sandstone and siltstone; and gray, tan, or green sandstone and mudstone (Roehler, 1991a, 1991b, 1992a, 1992b, 1992c)." (Bartos, et al., 2015).
Wilkins Peak confining unit	Sedimentary rock aquitard (consolidated or semi-consolidated rock)	" Confining units are the Wilkins Peak and Tipton confining units ." (Bartos, et al., 2015).
Farson Sandstone-Alkali Creek Aquifer	Clastic sedimentary rock aquifer (consolidated or semi-consolidated rock) (Groundwater in the sandstone aquifers is under unconfined (water table) and confined (artesian) conditions.)	" Aquifers , from top to bottom, are the Bridger, Laney, Farson Sandstone-Alkali Creek, and Wasatch-Fort Union aquifers (fig. 6)." (Bartos, et al., 2015). "The laterally equivalent Farson Sandstone Member of the Green River Formation consists of gray, tan, and brown sandstone; thin interbedded, gray shale and siltstone of lacustrine origin; and locally occurring conglomerate ." (Bartos, et al., 2015).
Tipton confining unit	Sedimentary rock aquitard (consolidated or semi-consolidated rock)	" Confining units are the Wilkins Peak and Tipton confining units .".

Formation name	Category	Quote
Wasatch zone of the Wasatch-Fort Union aquifer	Unconsolidated aquifer (Groundwater in the sandstone aquifers is under unconfined (water table) and confined (artesian) conditions.)	"Aquifers, from top to bottom, are the Bridger, Laney, Farson Sandstone-Alkali Creek, and Wasatch-Fort Union aquifers (fig. 6)." (Bartos, et al., 2015). "Wasatch Formation consists of interbedded freshwater-deposited brown, green, and gray sandstone, siltstone, mudstone, and shale; and locally conglomeratic lenses (Roehler, 1991a, 1992a, 1992b, 1992c)." (Bartos, et al., 2015).
Uinta Mountain	Endogenous bedrock	"Transmissivity also is less along the Uinta Mountains resulting in relatively steep hydraulic gradients compared to the central part of the basin." (Bartos, et al., 2015). "The Uinta Mountain group (Uinta quartzite of previous reports) a series of brick-red to purplish-red quartzite and sandstone beds of pre-Cambrian age, aggregating more than 12,000 feet in thickness, makes up the central mass of the range." (Bradley, 1936).

Bartos, T.T., Hallberg, L.L., Eddy-Miller, C.A. (2015). *Hydrogeology, groundwater levels, and generalized potentiometric-surface map of the Green River Basin lower Tertiary aquifer system, 2010–14, in the northern Green River structural basin* (No. 2015-5090). US Geological Survey.

Bradley, W. H. (1936). *Geomorphology of the north flank of the Uinta Mountains* (No. 185-I, pp. 163-199). United States Government Printing Office.

States West Water Resources Corporation (2001). *Green River Basin Water Planning Process. Wyoming Water Development Commission Basin Planning Program. (Final report, pp. 178)* accessed 5/17/2022 via https://waterplan.state.wy.us/plan/green/finalrept/finalrept_lores.pdf

3.64 Ozark Plateaus Aquifer System

Supplementary Fig. 222. Hydrogeologic cross section. 20 equally spaced transparent pink bars overlie the cross section; each shaded bar depicts the vertical offset from the land surface to the top of the uppermost confining unit or endogenous bedrock.

Supplementary Fig. 223. Vertical variations in the prevalence of wells that have been defined as tapping an unconfined or a confined aquifer by the USGS. The smaller squares represent 10 m depth intervals from the land surface to 100 m; the larger squares represent 20 m intervals from 100 m to 300 m below the land surface.

The Ozark Plateaus Aquifer System is located in southern Missouri, northwestern Arkansas, northeastern Oklahoma, and southeastern Kansas.

(i) A hydrogeologic cross section presented in Fig. 3 by Clark et al. (2019) shows thick sequences of carbonate rock with interbedded confining units. The St. Francois confining unit exists at deep depths (>300 m) and, in the northwest, a shallower and thin confining unit exists (the Ozark confining unit).

(ii) We analysed wells within the study area that the USGS has defined as either unconfined or confined. Most (>80%) wells at depths of 280-300 m and at depths exceeding 280 m are defined as tapping a confined aquifer.

Depth to confined conditions: 280-300 meters below land surface (based on (ii) above)

Reference: Clark, B. R., Duncan, L. L., Knierim, K. J. (2019). Groundwater availability in the Ozark Plateaus aquifer system. US Geological Survey Professional Paper 1854, 95 pp. Accessed February 13, 2021 from <https://pubs.er.usgs.gov/publication/pp1854>

The table below presents a series of published quotes (see quotation marks denoting text quoted from another publication, which is cited following the quotation marks with the full reference written in full below the table). The leftmost column lists a title of a hydrogeologic formation depicted in the cross section on the previous page. The rightmost column presents a quote from a hydrogeological study (see base of table for citation). The quote has been annotated with colored text to highlight how we categorized each layer (i.e., see categories in the center column in the table). Specifically: (i) blue text highlights portions of a quote that provide insights into the degree of consolidation of the formation, (ii) red text highlights portions of a quote that categorize the formation as an aquifer or an aquitard (i.e., higher versus lower permeability in the context of local hydrogeologic formations), and (iii) green text highlights portions of a quote that provide information about the lithology of the formation.

Supplementary Table 67. Hydrostratigraphy details for the Ozark Plateau aquifer system

Formation name	Category	Quote
Western Interior Plain confining unit	Sedimentary rock aquitard (consolidated or semi-consolidated rock)	"The Ozark system is regionally overlain by the Western Interior Plains confining system in the southern and western extents of the study area (fig. 1). The confining system is mostly composed of shale with lesser amounts of limestone and sandstone. " (Clark et al., 2019).
Springfield Plateau aquifer	Carbonate aquifer	"The uppermost aquifer of the Ozark system is the Springfield Plateau aquifer of Mississippian age, which consists of limestone with varying chert abundance and has a median thickness of 237 ft (Hays and others, 2016; Westerman and others, 2016a)." (Clark et al., 2019).
Ozark confining unit	Sedimentary rock aquitard (consolidated or semi-consolidated rock)	"The Ozark confining unit is relatively thin (median thickness is 42 ft; Westerman and others, 2016a) to absent in some areas, which permits hydraulic connection of the underlying Ozark aquifer and overlying Springfield Plateau aquifer (fig. 2). Lithology of the Ozark confining unit varies throughout the study area, but is generally composed of low-permeability limestone, sandstone, and shale units (Hays and others, 2016)." (Clark et al., 2019).
Middle Ozark aquifer	Carbonate aquifer	"The Ozark aquifer includes productive dolostone units of the lower Ozark aquifer (median thickness of 885 ft), denser and relatively lower permeability dolostones of the middle Ozark aquifer (median thickness of 416 ft), and the mixed lithology of limestone, dolostone, shale, and limited sandstone units of the upper Ozark aquifer (median thickness of 590 ft) (Hays and others, 2016; Imes and Emmett, 1994; Westerman and others, 2016a) (fig. 2)." (Clark et al., 2019)." (Clark et al., 2019). "The middle Ozark aquifer therefore serves as an important groundwater resource, despite lower permeability than some of the more karstified units such as the Springfield Plateau aquifer or the lower Ozark aquifer." (Clark et al., 2019).
Lower Ozark aquifer	Carbonate aquifer	"The lower Ozark aquifer is generally the most productive part of the Ozark aquifer owing to the enhanced secondary and tertiary porosity and permeability from karst formations (Hays and others, 2016)." (Clark et al., 2019).
St. Francois confining unit	Sedimentary rock aquitard (consolidated or semi-consolidated rock)	"The St. Francois aquifer is confined throughout much of its extent where overlain by the St. Francois confining unit . The St. Francois confining unit of Cambrian age has a median thickness of 228 ft (Westerman and others, 2016a) and is composed of low-permeability shale, siltstone,

Formation name	Category	Quote
		dolostone, and limestone (Hays and others, 2016)." (Clark et al., 2019).
St. Francois aquifer	Sedimentary rock aquifer (consolidated or semi-consolidated rock)	"The basal hydrogeologic unit of the Ozark system is the St. Francois aquifer of Cambrian age (fig. 2), which has a median thickness of 291 ft (Westerman and others, 2016b) and is composed of permeable sandstones and dolostones (Hays and others, 2016)." (Clark et al., 2019).
Upper Ozark aquifer	Carbonate aquifer	"The Ozark aquifer includes productive dolostone units of the lower Ozark aquifer (median thickness of 885 ft), denser and relatively lower permeability dolostones of the middle Ozark aquifer (median thickness of 416 ft), and the mixed lithology of limestone, dolostone, shale, and limited sandstone units of the upper Ozark aquifer (median thickness of 590 ft) (Hays and others, 2016; Imes and Emmett, 1994; Westerman and others, 2016a) (fig. 2)." (Clark et al., 2019).
Mississippi embayment	Unconsolidated aquifer	"The remainder of net groundwater outflow (171 Mgal/d) occurs through the unconsolidated units of the Mississippi embayment on the eastern margin of the study area (fig. 3B), including the McNairy-Nacatoch aquifer of Cretaceous-age and Tertiary-age units." (Clark et al., 2019). "Mississippi embayment remains an important question because of the high amount of groundwater use from alluvial aquifers and limitations in groundwater-flow models at model boundaries." (Clark et al., 2019).
Basement rock	Endogenous bedrock	"The bottom of the Ozark system is bounded by metamorphic and igneous rocks of Precambrian age (Basement confining unit) that underlie much of the midwestern aquifers in the central United States (Jorgensen and others, 1993). "The unit exhibits very low permeability owing to the igneous and metamorphic rocks that compose the basement complex (Jorgensen and others, 1993) and therefore was not explicitly modeled for this study." (Clark et al., 2019).

Clark, B.R., Duncan, L.L., Knierim, K.J. (2019). Groundwater availability in the Ozark Plateaus aquifer system: U.S. Geological Survey Professional Paper 1854, 82 p., <https://doi.org/10.3133/pp1854>

3.65 Pearl and Chattahoochee Aquifer System

Supplementary Fig. 224. Hydrogeologic cross section. 20 equally spaced transparent pink bars overlie the cross section; each shaded bar depicts the vertical offset from the land surface to the top of the uppermost confining unit or endogenous bedrock.

Supplementary Fig. 225. Vertical variations in the prevalence of wells that have been defined as tapping an unconfined or a confined aquifer by the USGS. The smaller squares represent 10 m depth intervals from the land surface to 100 m; the larger squares represent 20 m intervals from 100 m to 300 m below the land surface.

The Pearl and Chattahoochee Aquifer System is located in southwestern Georgia and southeastern Alabama.

(i) A hydrogeologic cross section presented in Fig. 3 by Kidd (1987) depicts a series of dipping sedimentary formations. Note that the cross section presented by Kidd (1987) lacks a

quantitative vertical axis; whereas the cross sections we consulted for the other n=73 aquifer systems have quantitative vertical axes. We scaled the vertical scale of the cross section by Kidd (1987) by matching the topography depicted in the figure with an elevation profile along the cross section path. We stress that this cross section is more uncertain than others; we did not use the cross section for this study area to estimate the depth to confined conditions (i.e., we use data source (ii) described below).

(ii) We analysed wells within the study area that the USGS has defined as either unconfined or confined. Most (>80%) wells at depths of 20-30 m and at depths exceeding 20 m are defined as tapping a confined aquifer.

Depth to confined conditions: 20-30 meters below land surface (based on (ii) above)

Reference: Kidd, R.E. (1987). *Geohydrology and susceptibility of major aquifers to surface contamination in Alabama: Area 9.* US Geological Survey Water-Resources Investigations Report 87-4187, 37 pp. Accessed June 13, 2022 via <https://pubs.usgs.gov/wri/1987/4187/report.pdf>

The table below presents a series of published quotes (see quotation marks denoting text quoted from another publication, which is cited following the quotation marks with the full reference written in full below the table). The leftmost column lists a title of a hydrogeologic formation depicted in the cross section on the previous page. The rightmost column presents a quote from a hydrogeological study (see base of table for citation). The quote has been annotated with colored text to highlight how we categorized each layer (i.e., see categories in the center column in the table). Specifically: (i) blue text highlights portions of a quote that provide insights into the degree of consolidation of the formation, (ii) red text highlights portions of a quote that categorize the formation as an aquifer or an aquitard (i.e., higher versus lower permeability in the context of local hydrogeologic formations), and (iii) green text highlights portions of a quote that provide information about the lithology of the formation.

Supplementary Table 68. Hydrostratigraphy details for the Pearl and Chattahoochee

Formation name	Category	Quote
Nanafalia Formation (Tnf)	Sedimentary rock aquitard (consolidated or semi-consolidated rock)	" Sand, varicolored, poorly-sorted, clay, grayish-green, silt and gravel " Table 2 (Kidd, 1987). "The Nanafalia is not a major aquifer, but the basal sand of the formation is the uppermost part of the Nanafalia-Clayton aquifer, which is a major aquifer in the study area. " Kidd, 1987).
Clayton Formation (Tc)	Unconsolidated aquifer	"In the outcrop areas, the lower part of the formation consists of 5 to 15 feet of fine- to coarse-grained sand that locally contains gravel, lignite and clay pebbles. The base of the bed generally has a thin layer of sandstone. In the subsurface the basal sand grades upward into sand overlain by sandy to clayey fossiliferous limestone that may be as much as 25 feet thick. The Clayton grades into sandy limestone in downdip areas. The Clayton Formation is included with the Nanafalia Formation as part of the Nanafalia-Clayton aquifer. The Nanafalia-Clayton aquifer is considered a major aquifer in the study area (table 2). The Clayton will yield about 10 gal/min in most areas south of its outcrop. Yields of 0.5 Mgal/d may be available in the central and southern parts of Pike and Harbour Counties from the limestone. " (Kidd, 1987).
Providence Sand (Kp)	Sedimentary rock aquifer (consolidated or semi-consolidated rock)	"The Providence consists of fine- to coarse-grained sand that is micaceous and carbonaceous; and laminated to thin-bedded silty clay and massive, lignitic and kaolinitic clay. " (Kidd, 1987). "The Providence Sand, the uppermost part of the Providence-Ripley aquifer, is a potential source of 0.5 Mgal/d per well in southern Pike and Harbour Counties (table 2). The Providence is tapped by wells that supply the towns of Brundidge, Goshen, Louisville, and Clio; and the South Alabama Electrical Co-op. " (Kidd, 1987).
Ripley Formation (Kr)	Sedimentary rock aquifer (consolidated or semi-consolidated rock)	"The Ripley ranges in thickness from 270 to 500 feet. The formation consists of fine- to coarse-grained sand, carbonaceous clay, sandy clay, and thin beds of gravelly sand and limestone. The sand beds in the Ripley Formation are part of the Providence-Ripley aquifer that is a major aquifer in the southern halves of Pike and Barbour Counties where yields of 0.5 to 1.0 Mgal/d per well may be obtained (table 2)." (Kidd, 1987).
Demopolis Chalk	Sedimentary rock aquitard (consolidated or	"The Demopolis overlies the Mooreville in the western part of the study area and the Blufftown Formation in central and eastern parts. It consists of sandy micaceous calcareous clay that grades eastward into sandy clay. The Demopolis is

Formation name	Category	Quote
	semi-consolidated rock	relatively impermeable and generally is not a source of water in the study area. " (Kidd, 1987).
Blufftown Formation (Kb)/ Mooreville Chalk (Km)	Sedimentary rock aquitard (consolidated or semi-consolidated rock)	"The Blufftown increases in thickness from about 30 feet in the western part of the study area to more than 500 feet in the eastern part. In the eastern part of the study area the lower part of the formation consists of about 200 feet of fine- to coarse-grained sand and sandy clay and the upper part consists of 200 to 300 feet of calcareous sandy clay containing some thin beds of sand and calcareous sandstone. The Blufftown Formation intertongues with the Mooreville Chalk in Macon, Bullock, and Russell Counties (fig. 3)." (Kidd, 1987). "The Blufftown is not considered a major aquifer" (Kidd, 1987). "The Mooreville consists of about 500 feet of silty chalk and calcareous clay interbedded with thin layers of limestone and calcareous sandstone. The Mooreville Chalk is relatively impermeable and is not a source of water in the study area." (Kidd, 1987).

Kidd, R.E. (1987). *Geohydrology and Susceptibility of Major Aquifers to Surface Contamination in Alabama, Area 9*. No. 87-4187. Department of the Interior, US Geological Survey, Accessed June 20, 2022 via <https://pubs.usgs.gov/wri/1987/4187/report.pdf>

3.66 Salinas Valley

Supplementary Fig. 226. Hydrogeologic cross section. 20 equally spaced transparent pink bars overlaid the cross section; each shaded bar depicts the vertical offset from the land surface to the top of the uppermost confining unit or endogenous bedrock.

The Salinas Valley is located in west-central California.

(i) A hydrogeologic cross section presented in Fig. 3 by Hall (1992) demonstrates widespread confining beds (e.g., the Salinas Valley Aquitard in the northwest of the valley). The median depth to the top of the uppermost confined unit is 31 meters (25th-75th percentile range is 9 meters to 49 meters below land surface; see pink transparent bars in cross section to the left).

(ii) USGS well data (defining if the well taps an unconfined or confined aquifer) are insufficient to evaluate the depths at which the aquifer system transitions from unconfined to confined conditions.

(iii) Hall (1992) states, with respect to confining conditions, (quote): "The blue clay, overlying the 180 foot aquifer, ranges from 25 feet thick at Salinas to more than 100 feet thick at Nashua Road. Is it known as the Salinas Aquitard and is composed mostly of blue marine clays with some silts". A report by Harding ESE (2001) on the hydrostratigraphy area states that (quote): "The lateral extent of the 180-Foot Aquifer is generally defined by the overlying [Salinas Valley Aquitard] clay, which maintains confined conditions throughout much of the study area." Vengosh et al. (2002) state that: "The Salinas Valley has a

relatively deep, confined "400-foot" aquifer, overlain by a "180-foot" aquifer and a shallower perched aquifer, all made up of alluvial sand, gravel and clay deposits"

Depth to confined conditions: 30-40 m based on (i) (i.e., median length of the 20 pink transparent bars overlaid on cross section to the left)

References: Hall, P. (1992). Selected geological cross sections in Salinas Valley using geobase. Monterey County Water Resources Agency Basin Management Plan Report, 68 pp. Accessed April 11, 2022 via https://digitalcommons.csumb.edu/cgi/viewcontent.cgi?article=1025&context=hornbeck_cgb_6_a

Harding ESE (2001). Final Report Hydrogeologic Investigation of the Salinas Valley Basin in the Vicinity of Fort Ord and Marina Salinas Valley, California. Report for the Monterey County Water Resources Agency, 165 pp. Accessed May 20, 2022 via <http://svbqsa.org/wp-content/uploads/2020/08/2001-Final-Report-Hydrog.pdf>

Vengosh, A., Gill, J., Davisson, M.L., Hudson, B.G. (2002). A multi-isotope (B, Sr, O, H, and C) and age dating (³H-³He and ¹⁴C) study of groundwater from Salinas Valley, California: Hydrochemistry, dynamics, and contamination processes. *Water Resources Research*, 38, 1008.

The table below presents a series of published quotes (see quotation marks denoting text quoted from another publication, which is cited following the quotation marks with the full reference written in full below the table). The leftmost column lists a title of a hydrogeologic formation depicted in the cross section on the previous page. The rightmost column presents a quote from a hydrogeological study (see base of table for citation). The quote has been annotated with colored text to highlight how we categorized each layer (i.e., see categories in the center column in the table). Specifically: (i) blue text highlights portions of a quote that provide insights into the degree of consolidation of the formation, (ii) red text highlights portions of a quote that categorize the formation as an aquifer or an aquitard (i.e., higher versus lower permeability in the context of local hydrogeologic formations), and (iii) green text highlights portions of a quote that provide information about the lithology of the formation.

Supplementary Table 69. Hydrostratigraphy details for the Salinas Valley

Formation name	Category	Quote
Recent Quaternary sediments	Unconfined aquifer	"The sands and gravels of this group supply most of the groundwater to the valley" (Hall, 1992). " Recent sand dunes, stream channel deposits, alluvium, floodplain deposits " (Hall, 1992). " low to high well yield " (Table 1. Hall, 1992).
Older alluvium	Sedimentary rock aquitard (consolidated or semi-consolidated rock)	"It is known as the Salinas aquitard and is composed mostly of blue marine clays with some silts . It has low permeability and greatly reduced river seepage and recharge to the 180 foot aquifer. The clay pinches out the East-side" (Hall, 1992). " Salinas aquiclude " (Hall, 1992).
Valley fill materials (salt water intrusion problem near the coast)	Clastic sedimentary rock aquifer (consolidated or semi-consolidated rock)	"There are fluvial sands and gravel associated with old Salinas River channel and also possible delta condition." (Hall, 1992). " Both the 180 foot aquifer and the upper part of the 400 foot aquifer may correlate with Aroma Sands . The lower part of the 400 foot aquifer may correlate with part of the Paso Robles formation ." (Hall, 1992).

Hall, P. (1992). Selected geological cross sections in Salinas Valley using geobase. Monterey County Water Resources Agency Basin Management Plan Report, 68 pp. Accessed April 11, 2022 via https://digitalcommons.csumb.edu/cqi/viewcontent.cqi?article=1025&context=hornbeck_cqb_6_a

3.67 Salt Lake Valley

Supplementary Fig. 227. Hydrogeologic cross section. 20 equally spaced transparent pink bars overlie the cross section; each shaded bar depicts the vertical offset from the land surface to the top of the uppermost confining unit or endogenous bedrock.

Supplementary Fig. 228. Vertical variations in the prevalence of wells that have been defined as tapping an unconfined or a confined aquifer by the USGS. The smaller squares represent 10 m depth intervals from the land surface to 100 m; the larger squares represent 20 m intervals from 100 m to 300 m below the land surface.

Salt Lake Valley is located in central Utah.

(i) A hydrogeologic cross section presented in Fig. 4 by Manning and Solomon (2005) labels a fine-grained confining layer at shallow depths (<100 m) across much of the study area.

(ii) We analysed wells within the study area that the USGS has defined as either unconfined or confined. Most (>80%) wells at depths of 50-60 m and at depths exceeding 50 m are defined as tapping a confined aquifer.

Depth to confined conditions: 50-60 m (based on (ii) above)

Reference: Manning, A.H., Solomon, D.K. (2005). An integrated environmental tracer approach to characterizing groundwater circulation in a mountain block. *Water Resources Research*, 41, W12412. doi:10.1029/2005WR004178

The table below presents a series of published quotes (see quotation marks denoting text quoted from another publication, which is cited following the quotation marks with the full reference written in full below the table). The leftmost column lists a title of a hydrogeologic formation depicted in the cross section on the previous page. The rightmost column presents a quote from a hydrogeological study (see base of table for citation). The quote has been annotated with colored text to highlight how we categorized each layer (i.e., see categories in the center column in the table). Specifically: (i) blue text highlights portions of a quote that provide insights into the degree of consolidation of the formation, (ii) red text highlights portions of a quote that categorize the formation as an aquifer or an aquitard (i.e., higher versus lower permeability in the context of local hydrogeologic formations), and (iii) green text highlights portions of a quote that provide information about the lithology of the formation.

Supplementary Table 70. Hydrostratigraphy details for the Salt Lake Valley

Formation name	Category	Quote
Shallow Unconfined aquifer and Fine-grained Confining Layer (this aquifer has thin layer of confining unit)	Unconsolidated aquifer	"Where it is confined, the principal aquifer is overlain by a shallow unconfined aquifer typically <30 m thick." (Manning & Solomon, 2005). "the principal aquifer is overlain by a layer of fine-grained sediments that act as a confining layer. " (Manning & Solomon, 2005). "The unconsolidated sediments of Quaternary age were deposited mainly as alluvial fans, by streams, and as deltas and other lacustrine features associated with Lake Bonneville and older paleolakes that once covered the valley. The hydraulic conductivity of coarser grained deposits is estimated to be about 200 ft/d, compared to a value of about 1 ft/d for shallow lake-deposited clays (Lambert, 1995, p. 14)." (Thiros et al., 2010)
Principal aquifer (Quaternary Fill)	Unconsolidated aquifer	" Production wells are screened within the principal aquifer, the deeper Quaternary sediments composed of sand and gravel interbedded with lenses of silt and clay. " (Manning & Solomon, 2005).
Consolidated to semi-consolidated Tertiary	Clastic sedimentary rock aquifer (consolidated or semi-consolidated rock)	"Quaternary sediments are 100–300 m thick and Tertiary sediments are 200–800 m thick throughout most of the basin. " (Manning & Solomon, 2005). "The Tertiary-age sediments that crop out along the western and southern margins of the valley were deposited mainly as alluvial fans, in lakes, and as volcanic ash and are estimated to have a hydraulic conductivity of about 1 ft/d (Lambert, 1995, p. 15)" (Thiros et al., 2010)
Consolidated (Intrusive, Precambrian and sedimentary rock can be found)	Endogenous bedrock	"The geology of the Wasatch mountain block varies considerably within the study area [Bryant, 1990] and has been summarized by Manning and Solomon [2004]. Dominant rock types include granitic intrusive rocks, quartzites interbedded with shales, and mixed sedimentary rocks. " (Manning & Solomon, 2005). "The transmissivity of these rocks is dependent on the presence or absence of fractures and is highly variable. Hely and others (1971, plate 1) characterized the volcanic rocks as " rocks of lowest permeability. " (Thiros et al., 2010)

Manning, A.H., Solomon, D.K. (2005). An integrated environmental tracer approach to characterizing groundwater circulation in a mountain block. *Water Resources Research*, 41(12). | Thiros, S. A., Bexfield, L. M., Anning, D. W., & Huntington, J. M. (2010). *Conceptual understanding and groundwater quality of selected basin-fill aquifers in the southwestern United States* (No. 1781). US Geological Survey. Accessed June 3, 2022 via <https://pubs.usgs.gov/pp/1781/>

3.68 San Pedro Basin

Supplementary Fig. 229. Hydrogeologic cross section. 20 equally spaced transparent pink bars overlie the cross section; each shaded bar depicts the vertical offset from the land surface to the top of the uppermost confining unit or endogenous bedrock.

Supplementary Fig. 230. Vertical variations in the prevalence of wells that have been defined as tapping an unconfined or a confined aquifer by the USGS. The smaller squares represent 10 m depth intervals from the land surface to 100 m; the larger squares represent 20 m intervals from 100 m to 300 m below the land surface.

The San Pedro Basin is located in southeastern Arizona.

(i) A hydrogeologic cross section presented in Fig. 2 by Hopkins et al. (2014) depicts a clay layer in the central portion of the basin.

(ii) We analysed wells within the study area that the USGS has defined as either unconfined or confined. Most (>80%) wells at depths of 300-320 m and at depths exceeding 300 m are defined as tapping a confined aquifer.

Depth to confined conditions: 300-320 m (based on (ii) above)

Reference: Hopkins, C. B., McIntosh, J. C., Eastoe, C., Dickinson, J. E., Meixner, T. (2014). Evaluation of the importance of clay confining units on groundwater flow in alluvial basins using solute and isotope tracers: the case of Middle San Pedro Basin in southeastern Arizona (USA). *Hydrogeology Journal*, 22, 829-849.

The table below presents a series of published quotes (see quotation marks denoting text quoted from another publication, which is cited following the quotation marks with the full reference written in full below the table). The leftmost column lists a title of a hydrogeologic formation depicted in the cross section on the previous page. The rightmost column presents a quote from a hydrogeological study (see base of table for citation). The quote has been annotated with colored text to highlight how we categorized each layer (i.e., see categories in the center column in the table). Specifically: (i) blue text highlights portions of a quote that provide insights into the degree of consolidation of the formation, (ii) red text highlights portions of a quote that categorize the formation as an aquifer or an aquitard (i.e., higher versus lower permeability in the context of local hydrogeologic formations), and (iii) green text highlights portions of a quote that provide information about the lithology of the formation.

Supplementary Table 71. Hydrostratigraphy details for the San Pedro Basin

Formation name	Category	Quote
Upper Basin Fill	Unconsolidated aquifer	" coarse granular fraction of sedimentary basin fill represented by gravels and sands. It corresponds to the more hydraulically conductive portions of the Upper- and Lower-Basin Fill." (Callegary et al., 2016).
Clay Layer	Clastic sedimentary rock aquifer (consolidated or semi-consolidated rock)	"the fine sediments with low hydraulic conductivity that mainly comprise the upper basin fill. These low-conductivity silts and clays occur mainly in the central portion of the basin." (Callegary et al., 2016).
Lower Basin Fill	Clastic sedimentary rock aquifer (consolidated or semi-consolidated rock)	" fractured-rock aquifers , among which are the conglomeratic units of the Báucarit Formation, the Tc unit (See Chapter 4), the Tertiary felsic volcanic rocks that lie between these, and the fractured or weathered portions of the basement, such as limestone , that could possibly contain groundwater ." (Callegary et al., 2016).
Bedrock	Endogenous bedrock (intense fracturing of the basement)	"the oldest rocks form a Precambrian basement characterized by the Pinal Schist (1680 million years before present (Ma)) and mesoproterozoic granitic intrusions " (Callegary et al., 2016). " Hydrostratigraphic Basement " (Callegary et al., 2016)

Callegary, J.B., Minjárez Sosa, I., Tapia Villaseñor, E.M., dos Santos, P., Monreal Saavedra, R., Grijalva Noriega, F.J., Huth, A.K., Gray, F., Scott, C.A., Megdal, S.B., Oroz Ramos, L.A., Rangel Medina, M., Leenhouts, J.M. (2016). San Pedro River Aquifer Binational Report: International Boundary and Water Commission, 170 pp. Accessed November 29, 2021 from https://ibwc.gov/Files/San_Pedro_River_Binational%20Report_013116.pdf

3.69 Santa Clara-Calleguas Basin

Supplementary Fig. 231. Hydrogeologic cross section. 20 equally spaced transparent pink bars overlie the cross section; each shaded bar depicts the vertical offset from the land surface to the top of the uppermost confining unit or endogenous bedrock.

Supplementary Fig. 232. Vertical variations in the prevalence of wells that have been defined as tapping an unconfined or a confined aquifer by the USGS. The smaller squares represent 10 m depth intervals from the land surface to 100 m; the larger squares represent 20 m intervals from 100 m to 300 m below the land surface.

The Santa Clara-Calleguas Basin (Oxnard Plain) is located in coastal southern California.

(i) A hydrogeologic cross section presented in Fig. 8 by Hanson et al. (2002) depicts a multi-layered aquifer system with a confining unit relatively close to the coast.

(ii) We analysed wells within the study area that the USGS has defined as either unconfined or confined. Most (>80%) wells at depths of 50-60 m and at depths exceeding 50 m are defined as tapping a confined aquifer.

Depth to confined conditions: 50-60 m (based on (ii) above)

Reference: Hanson, R.T., Martin, P., Kocot, K.M. (2002). Simulation of ground-water/surface-water flow in the Santa Clara-Calleguas ground-water basin, Ventura County, California. Water-Resources Investigations Report 2002-4136, 172 pp. Accessed March 20, 2021 from <https://pubs.usgs.gov/wri/wri024136/wri024136.pdf>

The table below presents a series of published quotes (see quotation marks denoting text quoted from another publication, which is cited following the quotation marks with the full reference written in full below the table). The leftmost column lists a title of a hydrogeologic formation depicted in the cross section on the previous page. The rightmost column presents a quote from a hydrogeological study (see base of table for citation). The quote has been annotated with colored text to highlight how we categorized each layer (i.e., see categories in the center column in the table). Specifically: (i) blue text highlights portions of a quote that provide insights into the degree of consolidation of the formation, (ii) red text highlights portions of a quote that categorize the formation as an aquifer or an aquitard (i.e., higher versus lower permeability in the context of local hydrogeologic formations), and (iii) green text highlights portions of a quote that provide information about the lithology of the formation.

Supplementary Table 72. Hydrostratigraphy details for the Santa Clara-Calleguas Basin

Formation name	Category	Quote
Shallow aquifer	Unconsolidated aquifer	"The unconsolidated deposits of the late Pleistocene and Holocene epochs are grouped into the regional upper-aquifer system , which includes the Shallow, Oxnard, and Muqu aquifers (fig. 7B)." (Hanson et al., 2002). "Along the flood plain of the Santa Clara River, the shallow aquifer consists of predominantly sand and gravel and is an important source of ground water ." (Hanson et al., 2002).
Oxnard aquifer	Unconsolidated aquifer	"The unconsolidated deposits of the late Pleistocene and Holocene epochs are grouped into the regional upper-aquifer system , which includes the Shallow, Oxnard, and Muqu aquifers (fig. 7B)." (Hanson et al., 2002). " The Oxnard aquifer lies at the base of the Holocene deposits and consists of sand and gravel deposited by the ancestral Santa Clara River and the Calleguas Creek and by their major tributaries ." (Hanson et al., 2002).
Muqu aquifer	Unconsolidated aquifer	"The unconsolidated deposits of the late Pleistocene and Holocene epochs are grouped into the regional upper-aquifer system , which includes the Shallow, Oxnard, and Muqu aquifers (fig. 7B)." (Hanson et al., 2002). "Throughout most of the ground-water basin, the Muqu aquifer extends from about 200 to 400 ft below land surface (fig. 8) and consists of sand and gravel interbedded with silt and clay ." (Hanson et al., 2002).
Upper Hueneme aquifer (Hueneme aquifer classification is based on electric log data)	Unconsolidated aquifer	"The lower-aquifer system is composed of complexly faulted and folded unconsolidated deposits of the Pliocene and Pleistocene epochs and include the upper and lower Hueneme, Fox Canyon, and Grimes Canyon aquifers (fig. 7B)." (Hanson et al., 2002). " These aquifers consist of lenticular layers of sand, gravel, silt, and clay. The sediments constituting the aquifers have been subjected to considerable folding, faulting, and erosion since deposition ." (Hanson et al., 2002).
Lower Hueneme aquifer (more fine-grained deposits in the Lower than upper Hueneme aquifer)	Unconsolidated aquifer	"The lower-aquifer system is composed of complexly faulted and folded unconsolidated deposits of the Pliocene and Pleistocene epochs and include the upper and lower Hueneme, Fox Canyon, and Grimes Canyon aquifers (fig. 7B)." (Hanson et al., 2002). " These aquifers consist of lenticular layers of sand, gravel, silt, and clay. The sediments constituting the aquifers have been subjected to considerable folding, faulting, and erosion since deposition ." (Hanson et al., 2002).

Formation name	Category	Quote
Fox Canyon aquifer	Unconsolidated aquifer	"The lower-aquifer system is composed of complexly faulted and folded unconsolidated deposits of the Pliocene and Pleistocene epochs and include the upper and lower Hueneme, Fox Canyon , and Grimes Canyon aquifers (fig. 7B)." (Hanson et al., 2002). "The aquifer consists of weakly indurated very fine- to medium-grained fossiliferous sand with occasional gravel and clay layers of shallow marine origin. " (Hanson et al., 2002).
Undifferentiated or non-water bearing rocks. (Includes Grime Canyon aquifers)	Sedimentary rock aquitard (consolidated or semi-consolidated rock)	"The Santa Barbara Formation (Weber and others, 1976), which consists of non-water-bearing marine sandstone, siltstone, mudstone, and shale, underlies the Fox Canyon aquifer throughout most of the ground-water basin and is considered the base of the ground-water system throughout most of the basin." (Hanson et al., 2002).

Hanson, R.T., Martin, P., Koczot, K.M. (2002). Simulation of ground-water/surface-water flow in the Santa Clara-Calleguas ground-water basin, Ventura County, California. Water-Resources Investigations Report 2002-4136, 172 pp. Accessed June 1, 2022 from <https://pubs.usgs.gov/wri/wri024136/wri024136.pdf>

3.70 Santa Rosa Valley

Supplementary Fig. 233. Hydrogeologic cross section. 20 equally spaced transparent pink bars overlie the cross section; each shaded bar depicts the vertical offset from the land surface to the top of the uppermost confining unit or endogenous bedrock.

The Santa Rosa Valley is located in western California.

(i) A hydrogeologic cross section presented in Fig. 2-9 by Santa Rosa Plain Advisory Panel (2014) suggests that the aquifer system does not contain a clear confining unit. The median depth to undifferentiated basement rock is 503 m (see transparent pink shaded bars on cross section; the 25th-75th percentile range of the depth to basement rock is 462-630 meters below land surface). However, the median depth to the top of the Petaluma Formation—which is depicted as low-permeability by Woolfenden and Nishikawa (2014) in their Fig. 5—is 111 m (25th-75th percentile range of the depth to the top of the Petaluma Formation is 68 to 132 meters below land surface).

(ii) There are no USGS wells within the study area that have been defined as tapping an aquifer that is either unconfined or confined.

(iii) Santa Rosa Plain Advisory Panel (2014) state that (following text quoted directly): “In most parts of the study area, shallow groundwater flow is unconfined, but, at depth, groundwater flow is confined.” Further, Woolfenden and Nishikawa (2014) states (quote): “The Petaluma Formation is dominated by fine-grained materials, either in thick beds or as interstitial material in poorly sorted silty and clayey sands or gravels.” (see also their

conceptual model in Fig. 5 by Woolfenden and Nishikawa (2014) depicting the Petaluma Formation as a low-permeability formation).

Depth to confined conditions: 100-120 meters below land surface (on the basis of (i))

References: Santa Rosa Plain Advisory Panel (2014). *Santa Rosa Plain Watershed Groundwater Management Plan*. 335 pp. Accessed April 10, 2022 via http://santarosaplainingroundwater.org/wp-content/uploads/SRP_GMP_12-14.pdf

Woolfenden, L.R., Nishikawa, T. (2014). *Simulation of groundwater and surface-water resources of the Santa Rosa Plain watershed, Sonoma County, California*. US Geological Survey Scientific Investigations Report 2014–5052, 292 pp. Accessed May 19, 2022 via <https://pubs.usgs.gov/sir/2014/5052/pdf/sir2014-5052.pdf>

The table below presents a series of published quotes (see quotation marks denoting text quoted from another publication, which is cited following the quotation marks with the full reference written in full below the table). The leftmost column lists a title of a hydrogeologic formation depicted in the cross section on the previous page. The rightmost column presents a quote from a hydrogeological study (see base of table for citation). The quote has been annotated with colored text to highlight how we categorized each layer (i.e., see categories in the center column in the table). Specifically: (i) blue text highlights portions of a quote that provide insights into the degree of consolidation of the formation, (ii) red text highlights portions of a quote that categorize the formation as an aquifer or an aquitard (i.e., higher versus lower permeability in the context of local hydrogeologic formations), and (iii) green text highlights portions of a quote that provide information about the lithology of the formation.

Supplementary Table 73. Hydrostratigraphy details for the Santa Rosa Plain

Formation name	Category	Quote
Quaternary deposits and Glen Ellen Formation, undivided	Unconsolidated aquifer	"Overlying the basement rocks are five geologic units of Cenozoic age that form the SRP's primary aquifers . These are: (1) Quaternary Alluvium, (2) Glen Ellen Formation, (3) Wilson Grove Formation, (4) Petaluma Formation and (5) Sonoma Volcanics. " (Santa Rosa Plain Advisory Panel 2014). "These deposits are dominated by alluvial fan sediment deposits , which are materials eroded from rock exposed in the flanking hills. The deposits generally consist of mixed poorly- to well-sorted sand, silt, clay, gravel, cobbles and boulders, as interfingering, variably thin or thick beds of limited lateral extent (tens to hundreds of feet). " (Santa Rosa Plain Advisory Panel 2014). The Glen Ellen Formation consists of clay-rich stratified stream deposits of poorly sorted sand, silt, and gravel (Table 2-4). Beds of these sediments vary from coarse to fine-grained, commonly over distances of a few tens to a few hundreds of feet, both laterally and vertically. (Santa Rosa Plain Advisory Panel 2014)
Wilson Grove Formation	Unconsolidated aquifer	" The Wilson Grove Formation is relatively thick (300 ft to greater than 1000 ft thick), and mostly composed of weakly cemented marine-deposited sandstone, with volcanic ash intervals. The predominance of relatively clean sand and the low degree of cementation in the Wilson Grove Formation result in moderate to high permeability. Well production in the Wilson Grove Formation is high: from 200 to 1,000 gpm or more. " (Santa Rosa Plain Advisory Panel 2014).
Sonoma Volcanics	Volcanic aquifer	"These rocks comprise a highly variable assemblage of andesitic and basaltic tuffs with interbedded lava flows and explosive volcanoclastic rocks, having a broad range of water-bearing properties. " (Santa Rosa Plain Advisory Panel 2014). " Water production from wells drilled into thick air-fall pumice units may exceed a few hundred gpm, but wells drawing from unfractured lavas or welded tuffs may produce less than 10 gpm and dry holes are encountered occasionally. " (Santa Rosa Plain Advisory Panel 2014)
Petaluma Formation	Clastic sedimentary rock aquitard (consolidated or	"The Petaluma Formation is dominated by more or less consolidated silt or clay-rich mudstone, with local beds and lenses of poorly-sorted sandstone and minor conglomerate beds. Due to the large amount of silt- and

Formation name	Category	Quote
	semi-consolidated rock)	clay-sized particles, the specific yields of wells are low, varying from 3 to 7 percent. Domestic wells drilled into the Petaluma Formation yield on average about 20 gpm and vary from 10 to 50 gpm. (Santa Rosa Plain Advisory Panel 2014).
Undifferentiated basement	Endogenous bedrock	“Sandstone, greywacke, chert, serpentine.” (Santa Rosa Plain Advisory Panel 2014). “Basement rock”. (Santa Rosa Plain Advisory Panel 2014). “While the basement rocks provide a viable, sole source supply for many households, they are not considered a major water supply source in the SRP groundwater subbasin.”. (Santa Rosa Plain Advisory Panel 2014).

Santa Rosa Plain Advisory Panel (2014). Santa Rosa Plain Watershed Groundwater Management Plan. 335 pp. Accessed 5/17/22 (http://santarosaplaingroundwater.org/wp-content/uploads/SRP_GMP_12-14.pdf)

3.71 South Park Basin

Supplementary Fig. 234. Hydrogeologic cross section. 20 equally spaced transparent pink bars overlie the cross section; each shaded bar depicts the vertical offset from the land surface to the top of the uppermost confining unit or endogenous bedrock.

Supplementary Fig. 235. Vertical variations in the prevalence of wells that have been defined as tapping an unconfined or a confined aquifer by the USGS. The smaller squares represent 10 m depth intervals from the land surface to 100 m; the larger squares represent 20 m intervals from 100 m to 300 m below the land surface.

The South Park Basin is located in central Colorado.

(i) A hydrogeologic cross section presented in Fig. 11 by Barkmann et al. (2013) demonstrates the complexity of the sedimentary rock sequences in the South Park Basin. Among the 20 evenly spaced pink transparent bars (the length of each representing the depth to the uppermost confining unit), the median depth to a confining unit is >576 meters below land surface (25th-75th percentile range: 224 m to >619 m).

(ii) We analysed wells within the study area that the USGS has defined as either unconfined or confined. All wells (n=52) in the South Park Basin (depths of 0.15 m to 183 m) are classified as unconfined.

Depth to confined conditions: >576 m (based on (i), consistent with data presented in (ii))

References: Barkmann, P.E., Moore, A., Johnson, J. (2013) South Park Groundwater Quality Scoping Study. Report for Coalition for the Upper South Platte, 74 pp. Accessed November 29, 2021 from <https://cusp.ws/wp-content/uploads/2014/10/South-Park-Groundwater-Quality-Scoping-Study-Final.pdf>

The table below presents a series of published quotes (see quotation marks denoting text quoted from another publication, which is cited following the quotation marks with the full reference written in full below the table). The leftmost column lists a title of a hydrogeologic formation depicted in the cross section on the previous page. The rightmost column presents a quote from a hydrogeological study (see base of table for citation). The quote has been annotated with colored text to highlight how we categorized each layer (i.e., see categories in the center column in the table). Specifically: (i) blue text highlights portions of a quote that provide insights into the degree of consolidation of the formation, (ii) red text highlights portions of a quote that categorize the formation as an aquifer or an aquitard (i.e., higher versus lower permeability in the context of local hydrogeologic formations), and (iii) green text highlights portions of a quote that provide information about the lithology of the formation.

Supplementary Table 74. Hydrostratigraphy details for the South Park Basin

Formation name	Category	Quote
Conglomerate Member	Unconsolidated aquifer	"The Paleocene conglomerate member (informal name) is next and contains clasts of the older volcanic material mixed with clasts of limited Precambrian basement, Paleozoic rocks, and Cretaceous intrusives indicating a source from the Sawatch uplift to the west (Widmann and others, 2005; Kirkham and others, 2006)." (Barkmann et al., 2013). " lenticular conglomerate, sandstone, siltstone and mudstone " (Donegan, 2018). " Lower South Park Aquifer " (Donegan, 2018).
Laramie Formation	Sedimentary rock aquitard (consolidated or semi-consolidated rock) (some may form Laramie-Fox Hills Aquifer)	"Laramide hydrogeologic units consist primarily of various sedimentary members that collectively comprise the South Park aquifer and its confining units. " (Barkmann et al., 2013). " shale, sandstone, and coal " (Donegan, 2018). " confining unit " (Donegan, 2018).
Fox Hills Sandstone	Clastic sedimentary rock aquifer (consolidated or semi-consolidated rock)	"The Fox Hills Sandstone is in turn overlain by, and interfingers with, non-marine Laramie Formation. This upper unit of the group consists of overbank shale interbedded with lenticular beds of sandstone deposited on a low-relief coastal plain following the retreat of the Interior Seaway." (Barkmann et al., 2013). (Barkmann et al., 2013). " Potentially vulnerable aquifers within Cretaceous seaway hydrogeologic units include the Dakota and Laramie-Fox Hills aquifers " (Barkmann et al., 2013).
Pierre Shale	Sedimentary rock aquitard (consolidated or semi-consolidated rock)	" shale, sandstone, bentonitic layers " (Donegan, 2018). " confining unit, but sand beds and fractures can be a local aquifer " (Donegan, 2018).
Dakota Formation	Clastic sedimentary rock aquifer (consolidated or semi-consolidated rock)	"Conformably above the Morrison is the Cretaceous Dakota Formation which consists of sandstone, pebble conglomerate, and non-calcareous shale (Scarborough, 2001) deposited along the shoreline of the advancing Interior Seaway." (Barkmann et al., 2013). " Potentially vulnerable aquifers within Cretaceous seaway hydrogeologic units include the Dakota and Laramie-Fox Hills aquifers " (Barkmann et al., 2013).
Morrison Formation	Sedimentary rock aquitard (consolidated or	"The Morrison Formation consists of interbedded shale, sandstone, claystone, and basal limestone (Widmann and

Formation name	Category	Quote
	semi-consolidated rock	others, 2005)." (Barkmann et al., 2013). " confining unit, can yield water in porous zones " (Donegan, 2018).
Garó Formation	Unconsolidated aquifer	"The Garó Formation consists of calcareous sandstone and conglomerate that may conformably overlie the Maroon Formation (De Voto, 1971; Widmann and others, 2005)." (Barkmann et al., 2013). "Paleozoic hydrogeologic units in South Park include water-yielding Cambrian through Mississippian rocks, the Minturn-Maroon aquifer, and the Garó aquifer " (Barkmann et al., 2013).
Maroon Formation	Clastic sedimentary rock aquifer (consolidated or semi-consolidated rock) (Maroon Fm. Unit, variable, with porous zones interbedded with confining shales)	"The Maroon Formation overlies and is gradational with the Minturn Formation. It closely resembles the Minturn Formation in composition but has a redder color and contains less limestone . The change reflects a transition away from a marine-dominated clastic wedge to a sub-aerial fluvial clastic wedge (De Voto, 1971; Kirkham and others, 2006)." (Barkmann et al., 2013). "Paleozoic hydrogeologic units in South Park include water-yielding Cambrian through Mississippian rocks, the Minturn-Maroon aquifer , and the Garó aquifer" (Barkmann et al., 2013).
Pennsylvanian Minturn Formation	Clastic sedimentary rock aquifer (consolidated or semi-consolidated rock) (Minturn Fm. Unit, variable, with porous zones interbedded with confining shales)	"It grades upward into the coarser-grained Minturn Formation which contains interbedded pebble to cobble conglomerate, sandstone, siltstone, and limestone , attesting to intensified tectonic activity of the Anazasi uplifts." (Barkmann et al., 2013). "Paleozoic hydrogeologic units in South Park include water-yielding Cambrian through Mississippian rocks, the Minturn-Maroon aquifer , and the Garó aquifer" (Barkmann et al., 2013).
Pennsylvanian Minturn Formation with Evaporitic facies	Sedimentary rock aquitard (consolidated or semi-consolidated rock) (water quality issue when groundwater occurred)	"The Minturn Formation also includes an evaporitic facies that contains thick beds of salt and gypsum suggesting restricted circulation and high evaporation rates within the subsiding basin (Kirkham and others, 2007)" (Barkmann et al., 2013).
Igneous and metamorphic rocks	Endogenous bedrock	"The basin consists of eastward-dipping Paleozoic and Mesozoic sedimentary rocks preserved between uplifts of Precambrian igneous and metamorphic rocks on either side" (Barkmann et al., 2013). " Groundwater in Precambrian igneous and metamorphic rocks generally is upgradient from oil and gas sites" (Barkmann et al., 2013).

Donegan, K.C. (2018). Groundwater levels in the South Park Basin 2018. Colorado Division of Water Resources Report, 29 pp. Accessed November 29, 2021 from <https://dnrweblink.state.co.us/dwr/ElectronicFile.aspx?docid=3351305&dbid=0>

Barkmann, P.E., Moore, A., Johnson, J. (2013) South Park Groundwater Quality Scoping Study. Report for Coalition for the Upper South Platte, 74 pp. Accessed November 29, 2021 from https://cusp.ws/wp-content/uploads/2014/10/South-Park-Groundwater-Quality-Scoping-Study_Final.pdf

3.72 Tijuana-San Diego Basin

Supplementary Fig. 236. Hydrogeologic cross section. 20 equally spaced transparent pink bars overlie the cross section; each shaded bar depicts the vertical offset from the land surface to the top of the uppermost confining unit or endogenous bedrock.

Supplementary Fig. 237. Vertical variations in the prevalence of wells that have been defined as tapping an unconfined or a confined aquifer by the USGS. The smaller squares represent 10 m depth intervals from the land surface to 100 m; the larger squares represent 20 m intervals from 100 m to 300 m below the land surface.

The Tijuana-San Diego Basin is located in northwestern Baja California (Mexico) and southwestern California (US).

(i) A hydrogeologic cross section presented in a poster by Anders et al. (2012) depicts a series of sedimentary sequences including shallow Quaternary-aged alluvium underlain by Pliocene- to Eocene-aged formations.

(ii) We analysed wells within the study area that the USGS has defined as either unconfined or confined. Most (>80%) wells at depths of 140-160 m and at depths exceeding 140 m are defined as tapping a confined aquifer.

Depth to confined conditions: 140-160 m (based on (ii) above)

Reference: Anders, R., Cronquist, D.A., White, L.E., Danskin, W.R. (2012). Mapping saline groundwater in a coastal southern California Aquifer System. AGU Poster, 1 pp. Accessed April 11, 2022 via https://ca.water.usgs.gov/projects/sandiego/pdfs/AGU2012_poster.pdf

The table below presents a series of published quotes (see quotation marks denoting text quoted from another publication, which is cited following the quotation marks with the full reference written in full below the table). The leftmost column lists a title of a hydrogeologic formation depicted in the cross section on the previous page. The rightmost column presents a quote from a hydrogeological study (see base of table for citation). The quote has been annotated with colored text to highlight how we categorized each layer (i.e., see categories in the center column in the table). Specifically: (i) blue text highlights portions of a quote that provide insights into the degree of consolidation of the formation, (ii) red text highlights portions of a quote that categorize the formation as an aquifer or an aquitard (i.e., higher versus lower permeability in the context of local hydrogeologic formations), and (iii) green text highlights portions of a quote that provide information about the lithology of the formation.

Supplementary Table 75. Hydrostratigraphy details for the Tijuana-San Diego Basin

Formation name	Category	Quote
Quaternary alluvium	Unconsolidated aquifer	" Groundwater production comes primarily from shallow alluvial aquifers composed of Holocene-age sediments in numerous small basins (Fig 13, San Diego Formation of Pliocene age (Kennedy et al., 1975; Moore and Kennedy, 1975, Abbott, 1999) occurs predominantly south of Mission Valley and extends past the United States-Mexican border to locations beneath the city of Tijuana." (Lee & Normark, 2009)
Pliocene (SD Formation)	Clastic sedimentary rock aquifer (consolidated or semi-consolidated rock)	" Moderately permeable sedimentary formations , mostly of Pliocene and Eocene age, form a narrow coastal aquifer extending from coastal reservoirs to the Pacific Ocean (fig. 1)." (Lee & Normark, 2009) " San Diego Formation. This unit consists of Pliocene age well-sorted, medium to coarse sand, silty and clayey sand, sandy silt, and sandy clay (Huntley and others 1996)." Hydrologic Region South Coast, 2004). " The water-bearing units in the basin are the San Diego Formation (SDCWA, 1997) and Quaternary age alluvium." Hydrologic Region South Coast, 2004)
Oligocene (Otay Formation)	Clastic sedimentary rock aquifer (consolidated or semi-consolidated rock)	" groundwater in the Otay Formation and the Eocene layer as between 25,000 and 45,000 years before present." (Anders et al., 2012). "The Otay Formation consists predominantly of 35 to 50 m of white, volcanically derived tuffaceous fine sandstone with thin bentonitic interbeds, marked by a basal breccia conglomerate unit. " (Walsh and Demere 1991 quoting Artim and Pinckey (1973))
Eocene sedimentary rock	Clastic sedimentary rock aquifer (consolidated or semi-consolidated rock)	" Moderately permeable sedimentary formations , mostly of Pliocene and Eocene age, form a narrow coastal aquifer extending from coastal reservoirs to the Pacific Ocean (fig. 1)." (Lee & Normark, 2009)
Cretaceous crystalline rock	Endogenous bedrock	"The basin is surrounded by contacts with semi-permeable rocks of the Eocene Poway Group, impermeable Cretaceous crystalline rock, and impermeable Jurassic to Cretaceous Santiago Peak volcanic rocks. "

Lee, H.J., Normark, W.R. (2009). Earth science in the urban ocean: The Southern California continental borderland (Vol. 454). Geological Society of America.

Anders, R., Cronquist, D. A., White, L. E., & Danskin, W. R. (2012, December). Mapping Saline Groundwater in a Coastal Southern California Aquifer System. In *AGU Fall Meeting Abstracts* (Vol. 2012, pp. H53B-1524). | Walsh, Stephen L., and Thomas A. Demere (1991). Age and stratigraphy of the Sweetwater and Otay formations, San Diego County, California. 131-148.

[Hydrologic Region South Coast \(2004\). Tijuana Groundwater Basin. California's Groundwater Bulletin 118. Accessed 5/26/2022 via https://water.ca.gov/-/media/DWR-Website/Web-Pages/Programs/Groundwater-Management/Bulletin-118/Files/2003-Basin-Descriptions/9_019_Tijuana.pdf](https://water.ca.gov/-/media/DWR-Website/Web-Pages/Programs/Groundwater-Management/Bulletin-118/Files/2003-Basin-Descriptions/9_019_Tijuana.pdf)

[Hydrologic Region South Coast \(2004\). San Diego River Valley Groundwater Basin. California's Groundwater Bulletin 118. Accessed 5/26/2022 via https://water.ca.gov/-/media/DWR-Website/Web-Pages/Programs/Groundwater-Management/Bulletin-118/Files/2003-Basin-Descriptions/9_015_SanDiegoRiverValley.pdf](https://water.ca.gov/-/media/DWR-Website/Web-Pages/Programs/Groundwater-Management/Bulletin-118/Files/2003-Basin-Descriptions/9_015_SanDiegoRiverValley.pdf)

[Walsh, S. L., Demere, T. A. \(1991\). Age and stratigraphy of the Sweetwater and Otay formations, San Diego County, California \(https://archives.datapages.com/data/pac_sepm/086/086001/pdfs/131.htm\)](https://archives.datapages.com/data/pac_sepm/086/086001/pdfs/131.htm)

Artim, E. R., Pinckney, C. J. (1973). La Nacion Fault System, San Diego, California. Geological Society of America Bulletin, 84, 1075-1080.

3.73 Upper Santa Ana Basin

Supplementary Fig. 238. Hydrogeologic cross section. 20 equally spaced transparent pink bars overlie the cross section; each shaded bar depicts the vertical offset from the land surface to the top of the uppermost confining unit or endogenous bedrock.

Supplementary Fig. 239. Vertical variations in the prevalence of wells that have been defined as tapping a confined or an unconfined aquifer by the USGS. The smaller squares represent 10 m depth intervals from the land surface to 100 m; the larger squares represent 20 m intervals from 100 m to 300 m below the land surface.

The Upper Santa Ana Basin is located in southern California.

(i) A hydrogeologic cross section presented in Fig. 2-11 by Wildermuth (2005) does not depict a clear continuous confining unit in the Upper Santa Ana Basin.

(ii) We could not identify a depth where most (>80%) wells with the depth range and at deeper depths are defined as tapping a confined aquifer. The three deepest wells in our dataset (depths of 330 m, 338 m, 360 m) are classified as unconfined.

Depth to confined conditions: >360 m (based on (ii) above)

Reference: Wildermuth Environmental (2005). Chino Basin Optimum Basin Management Program State of the Basin Report - 2004. Report prepared for Chino Basin Watermaster. 643 pp. Accessed April 12, 2022 via http://www.cbwm.org/docs/engd/ocs/isob/ISOB_Final_FullVersion.pdf

The table below presents a series of published quotes (see quotation marks denoting text quoted from another publication, which is cited following the quotation marks with the full reference written in full below the table). The leftmost column lists a title of a hydrogeologic formation depicted in the cross section on the previous page. The rightmost column presents a quote from a hydrogeological study (see base of table for citation). The quote has been annotated with colored text to highlight how we categorized each layer (i.e., see categories in the center column in the table). Specifically: (i) blue text highlights portions of a quote that provide insights into the degree of consolidation of the formation, (ii) red text highlights portions of a quote that categorize the formation as an aquifer or an aquitard (i.e., higher versus lower permeability in the context of local hydrogeologic formations), and (iii) green text highlights portions of a quote that provide information about the lithology of the formation.

Supplementary Table 76. Hydrostratigraphy details for the Upper Santa Ana Basin

Formation name	Category	Quote
Hydrostratigraphic Unit: One	Unconsolidated aquifer	"Layer 1 consists of the upper 200-300 feet of sediments , and is generally representative of the shallow aquifer system (see Section 2.3.2.3). Layer 1 sediments are typically coarse-grained (sand and gravel layers) and, where saturated, transmit large quantities of groundwater to wells due to high hydraulic conductivities. " (Wildermuth Environmental, 2005).
Hydrostratigraphic Unit: Two	Clastic sedimentary rock aquifer (consolidated or semi-consolidated rock)	"Layer 2 consists of 200-500 feet of sediments underlying Layer 1, and is representative of the upper portion of the deep aquifer system (see Section 2.3.2.3). On the west side of Chino Basin, Layer 2 sediments are primarily fine-grained (silt and clay layers) with few interbedded sand and gravel layers. " (Wildermuth Environmental, 2005).
Hydrostratigraphic Unit: Three	Clastic sedimentary rock aquifer (consolidated or semi-consolidated rock)	"Layer 3 consists of 100-500 feet of sediments underlying Layer 2, and is representative of the lower portion of the deep aquifer system (see Section 2.3.2.3). Layer 3 sediments are confined to the deepest (central) portions of Chino Basin, and pinch-out toward the basin margins. Layer 3 sediments are typically coarse-grained (sand and gravel layers) , but due to their greater age, consolidation, and state of weathering, these sediments have lower permeability than the coarse-grained sediments of Layer 1. " (Wildermuth Environmental, 2005).
Sedimentary Bedrock	Sedimentary rock aquitard (consolidated or semi-consolidated rock)	"The base of the water-bearing sediments in this area occurs within the sedimentary bedrock formations that overlie the basement complex, and is recognized as a vertical transition to very low permeability sediments. " (Wildermuth Environmental, 2005). "Where encountered, the top of the black clays are interpreted as the bottom of the aquifer-system." (Wildermuth Environmental, 2005).
Crystalline Bedrock	Endogenous bedrock	"The basement complex consists of deformed and re-crystallized metamorphic rocks that have been invaded and displaced in places by huge masses of granitic and related igneous rocks. " (Wildermuth Environmental, 2005).

Wildermuth Environmental (2005). Chino Basin Optimum Basin Management Program State of the Basin Report - 2004. Report prepared for Chino Basin Watermaster. 643 pp. Accessed April 12, 2022 via http://www.cbwm.org/docs/engdocs/isob/ISOB_Final_FullVersion.pdf

3.74 Utah Lake Valley

Supplementary Fig. 240. Hydrogeologic cross section. 20 equally spaced transparent pink bars overlie the cross section; each shaded bar depicts the vertical offset from the land surface to the top of the uppermost confining unit or endogenous bedrock.

Supplementary Fig. 241. Vertical variations in the prevalence of wells that have been defined as tapping an unconfined or a confined aquifer by the USGS. The smaller squares represent 10 m depth intervals from the land surface to 100 m; the larger squares represent 20 m intervals from 100 m to 300 m below the land surface.

Utah Lake Valley is located in central Utah.

(i) A hydrogeologic cross section presented in Fig. 47 by Cederberg et al. (2009) depicts a series of shallow (i.e., near-surface) confining units.

(ii) We analysed wells within the study area that the USGS has defined as either unconfined or confined. Most (>80%) wells at depths of 10-20 m and at depths exceeding 10 m are defined as tapping a confined aquifer.

Depth to confined conditions: 10-20 m (based on (ii) above)

Reference: Cederberg, J.R., Gardner, P.M., Thiros, S.A., (2009). Hydrology of Northern Utah Valley, Utah County, Utah, 1975–2005. US Geological Survey Scientific Investigations Report 2008–5197. 114 pp. Accessed June 20, 2022 via <https://pubs.usgs.gov/sir/2008/5197/pdf/sir2008-5197.pdf>

Supplementary Table 77. Hydrostratigraphy details for the Utah Lake Valley

Formation name	Category	Quote
Overburden	Sedimentary rock aquitard (consolidated or semi-consolidated rock)	"all material above the first occurrence of a clay layer or above the water table in the pre-Lake Bonneville deposits was classified as overburden ."
Confining unit 1	Sedimentary rock aquitard (consolidated or semi-consolidated rock)	"It underlies the uppermost confining unit (CF1), a blue clay layer that is the most distinguishable and continuous layer identifiable in drillers' logs. This confining unit ranges in thickness from 50 to 150 ft. "
Shallow Pleistocene aquifer	Unconsolidated aquifer	"Hunt and others (1953) originally distinguished three confined aquifers in the basin-fill deposits of northern Utah Valley based on their relative depths and suspected age of sediment deposition associated with each aquifer. The naming convention applied with depth is: the shallow confined aquifer in deposits of Pleistocene age (SP aquifer), the deep confined aquifer in deposits of Pleistocene age (DP aquifer), and the confined aquifer in deposits of Quaternary/Tertiary age (QT aquifer). " (Cederberg et al., 2009). "The SP aquifer is the shallowest confined aquifer in northern Utah Valley and occurs throughout the middle and lower parts of the basin (fig. 10). The aquifer generally consists of deposits of silt, sand, and gravel with a thickness ranging from 10 to 150 ft. " (Cederberg et al., 2009).
Confining unit 2	Sedimentary rock aquitard (consolidated or semi-consolidated rock)	"Underlying the SP aquifer is the next major confining clay layer (CF2) that is often described in drillers' logs as blue or tan clay with a highly variable thickness ranging from 10 to 200 ft. " (Cederberg et al., 2009).
Deep Pleistocene aquifer	Unconsolidated aquifer	"The DP aquifer is highly variable in thickness and locally may consist of multiple coarser material layers separated by thin clay and silt layers. The aquifer consists of mixed sands and gravels close to the mountain front and grades to silty sand near the valley lowlands. " (Cederberg et al., 2009).
Confining unit 3	Sedimentary rock aquitard (consolidated or semi-consolidated rock)	"Often, wells do not fully penetrate the DP aquifer and thicknesses listed on drillers' logs may not represent the full thickness of the aquifer. When penetrated, the underlying clays forming the lowest confining unit (CF3) are described as white clays, hardpan, or conglomerate. " (Cederberg et al., 2009).
Pre-Lake Bonneville & Quaternary /Tertiary aquifers	Unconsolidated aquifer	"The QT aquifer is the least penetrated, developed, and documented aquifer in the basin-fill deposits of northern Utah Valley. Deposits are often described in drillers' logs as coarse-grained gravels and sands interbedded with clays and silts that are not correlated among wells and are assumed to be discontinuous." (Cederberg et al., 2009). "The unconsolidated sediments west of the Jordan River and Utah Lake are described in drillers' logs much the same as the QT aquifer deposits are described, consisting of coarse gravels and sands interbedded with clays and silts. " (Cederberg et al., 2009)

Cederberg, J.R., Gardner, P.M., Thiros, S.A. (2009). Hydrology of Northern Utah Valley, Utah County, Utah, 1975–2005: U.S. Geological Survey Scientific Investigations Report 2008–5197, 114 p. Accessed June 4, 2022 via <https://pubs.er.usgs.gov/publication/sir20085197>

Supplementary Note 4. Multiple rank correlations to account for interrelationships among our two potential explanatory variables

Here we report correlation coefficients between the depth below which modern groundwaters become groundwater becomes scarce, and (i) groundwater withdrawals, or (ii) the depth to confined conditions (table below on this page). After removing these N=17 aquifer systems, our Further, because (i) and (ii) may be statistically interrelated (i.e., it is possible that annual groundwater withdrawals are higher in aquifer systems where the depth to confined conditions is deeper), we complete multiple regression on the rank transforms our variables to account for these potential interrelationships and show that the partial regression coefficient between annual groundwater withdrawals and the depth that modern groundwater reaches remains statistically significant (Spearman P-value < 0.05; see table on final page of this Supplementary Note).

Supplementary Table 78. Spearman rank correlation coefficients (ρ) describing the statistical relationship between annual groundwater withdrawals (mm/year) and the depth below which most* samples contain minimal (<25%) modern water remains significant at water' and two potential explanatory variables. Significant (Spearman P-value < 0.05) correlation coefficients are in bold

Calculated depth below which most* samples contain minimal (<25%) modern water	Average groundwater pumping estimated for the year 2015 (mm/year)	Depth to confined conditions (meters below land surface)**
most* = 60%	$\rho = 0.39$ (P = 0.0006)	$\rho = 0.26$ (P = 0.0207) ***
most* = 70%	$\rho = 0.42$ (P = 0.0004)	$\rho = 0.42$ (P = 0.0002) ***
most* = 80%	$\rho = 0.40$ (P = 0.0014)	$\rho = 0.47$ (P = 0.0001) ***

* "most" interpreted as 60% (row 2 of table), 70% (row 3 of table) or 80% (row 4 of table)

** see Supplementary Note 3 for details describing how the depth to confined conditions was calculated.

*** For N=14 of the N=74 study aquifers, we could not identify a range of depths that the aquifer transitioned to confined conditions but, instead, we could confirm that any transition to confined conditions occurs at depths exceeding a given depth (for example, for the Southern High Plains the depth to confined conditions was found to be >270 m). To avoid excluding these aquifers in our rank correlation calculation (as a value of ">" does not readily lend itself to rank correlation), we ascribed the highest rank to these aquifers (i.e., all three were given an equal rank that exceeds that of all other aquifers for which we did identify a depth to confined conditions). In an effort to be transparent with respect to the approach we applied to incorporate values of "most": (i) 60% of samples greater than" in our rank regression, we created the figure on the following page showing the ranked values of "Depth to confined conditions" (see numbers along base of plot)

Formatted: Font: Times New Roman, English (United Kingdom)

Formatted: Normal

Formatted: Font: Times New Roman, English (United Kingdom)

Formatted: Font: Times New Roman, English (United Kingdom)

Formatted: Font: Times New Roman, Not Bold, English (United Kingdom)

Formatted: Font: Times New Roman, English (United Kingdom)

Formatted: Font: Times New Roman, English (United Kingdom)

Formatted: Font: Times New Roman, English (United Kingdom)

Formatted: Font: Times New Roman, English (United Kingdom)

Formatted: Font: Times New Roman, Font color: Custom Color(17,85,204), English (United Kingdom)

Formatted: Font: Times New Roman, English (United Kingdom)

Formatted: Font: Times New Roman, English (United Kingdom)

Formatted: Font: 10 pt

Formatted: Font: Times New Roman, 10 pt, Italic, Font color: Background 1, English (United Kingdom)

Formatted: Font: Times New Roman, 10 pt, Italic, Font color: Background 1, English (United Kingdom)

Formatted: Font: Times New Roman, 10 pt, Italic, Font color: Background 1, English (United Kingdom)

Supplementary Fig. 242. The depth to confined conditions for each of our N=74 study aquifers ranked from lowest (shallowest depth below land surface to confined conditions) to highest (deepest depth below the land surface to confined conditions). Each blue square represents one aquifer system (aquifer system titles shown along the bottom of the plot). Blue squares represent our estimate of the depth below the land surface to confined conditions, calculated on the basis of USGS-defined well conditions, hydrogeologic cross sections in local-scale reports, and quotes within local-scale reports (see Supplementary Note 3). The grey bars on the right side of the plot distinguish aquifer systems for which we could not identify a depth below which the aquifer system was confined; instead, for these aquifer systems, we identified a minimum depth to confined conditions (depicted by the blue square for these aquifer systems). The grey bars highlight that the true depth below which the aquifer system is mostly confined may be at any depth deeper wells (≠) than the blue square (i.e., any depth within the grey bar range). These aquifer systems were given the same (high) value within our rank regression including 'depth to confined conditions' (see table on previous page). The relative rank values (from lowest to highest) for the depth to confined conditions are shown as numbers along the bottom of the plot.

Formatted: Font: Bold, English (United Kingdom)

Formatted: English (United Kingdom)

Our analyses demonstrate significant (Spearman P-value < 0.05, and in most cases: Spearman P-value < 0.01) rank regressions of the depth below which modern groundwater is scarce versus (A) annual groundwater withdrawals, and (B) depth to confined conditions.

However, because the depth to confined conditions also weakly correlates with annual groundwater withdrawals (Spearman rank correlation coefficient of 0.16, Spearman P-value=0.17), we completed another statistical analysis to account for this interrelationship between our two explanatory variables (i.e., depth to confined conditions, and, annual groundwater withdrawals).

Specifically, we completed multiple regression on the rank transforms of depth to confined conditions and annual groundwater withdrawals. Our resulting partial regression coefficients (β values; table below) demonstrate that annual groundwater withdrawals is significantly (Spearman P-value < 0.05) correlated with the depth below which modern groundwater becomes scarce across all of our cut-off values (that is, 60%, 70% of samples from deeper wells, and (iii) or 80% of values used to determine the depth below which most samples from deeper wells; see Supplementary Table 10). These results are consistent with those arising when we include the N=17 aquifer systems in Supplementary Table 9 (see Supplementary Table 2); contain minimal (<25%) modern water; see second column in table below).

Supplementary Table 79. Multiple regression of the rank transforms coefficients (ρ) describing statistical relationship between ‘the depth below which most* samples contain minimal (<25%) modern water’ and our two potential explanatory variables: depth to confined conditions, and annual groundwater withdrawals. Significant (P-value < 0.05) partial regression coefficients (β) are in **bold**

Calculated depth below which most* samples contain minimal (<25%) modern water	Average groundwater pumping estimated for the year 2015 (mm/year)	Depth to confined conditions (meters below land surface) **
most* = 60%	$\beta = 0.34$ (P = 0.0027)	$\beta = 0.18$ (P = 0.0968)
most* = 70%	$\beta = 0.31$ (P = 0.0070)	$\beta = 0.32$ (P = 0.0036)
most* = 80%	$\beta = 0.29$ (P = 0.0144)	$\beta = 0.39$ (P = 0.0011)

* “most” interpreted as 60% (row 2 of table), 70% (row 3 of table) or 80% (row 4 of table)

Formatted: Normal

Formatted: Font: Times New Roman, English (United Kingdom)

Formatted: Font: Times New Roman, English (United Kingdom)

Formatted: Font: Times New Roman, Italic, English (United Kingdom)

Formatted: Font: Times New Roman, English (United Kingdom)

Formatted: Font: Times New Roman, Not Bold, English (United Kingdom)

Formatted: Font: 10 pt

Supplementary Note 5. Rank correlations between groundwater withdrawals and the depth below which modern groundwater is scarce if we exclude aquifer systems with shallow depths to confined conditions

We calculated statistical relationships between groundwater withdrawals and the depth below which modern groundwater is scarce in the case where we exclude aquifer systems with shallow confining units. Specifically, we re-ran our correlations after excluding all aquifer systems where the depth to confined conditions was found to be shallower than 50 m, or shallower than 100 m (see Supplementary Tables below). All rank correlations describing the statistical relationship between groundwater withdrawals and the depth below which most groundwater samples contain minimal modern groundwater remain statistically significant (Spearman P-value < 0.05) and have a positive sign (i.e., Spearman ρ values exceeding zero).

Supplementary Table 80. Spearman rank correlation coefficients (ρ) describing statistical relationship between ‘the depth below which most* samples contain minimal (<25%) modern water’ and two potential explanatory variables after excluding aquifer systems where the depth to confined conditions was estimated as shallower than 50 m. Significant (Spearman P-value < 0.05) correlations coefficients are in bold

Calculated depth below which most* samples contain minimal (<25%) modern water	Average precipitation divided by potential evapotranspiration (unitless)	Average topographic slope (percent grade)	Average groundwater pumping estimated for the year 2015 (mm/year)	Median thickness of permeable layers above bedrock (m)
most* = 60%	$\rho = -0.26$ (P = 0.025)	$\rho = 0.238$	$\rho = 0.1440$ (P = 0.0004); sample size of N=51	$\rho = 0.32$ (P = 0.006)
most* = 70%	$\rho = -0.33$ (P = 0.005)	$\rho = 0.146$	$\rho = 0.1946$ (P = 0.0010); sample size of N=48	$\rho = 0.33$ (P = 0.006)
most* = 80%	$\rho = -0.44$ (P = 0.0004)	$\rho = 0.047$	$\rho = 0.3145$ (P = 0.0027); sample size of N=41	$\rho = 0.36$ (P = 0.004)

* “most” interpreted as 60% (row 2 of table), 70% (row 3 of table) or 80% (row 4 of table)

Supplementary Table 81. Spearman rank correlation coefficients (ρ) describing statistical relationship between ‘the depth below which most* samples contain minimal (<25%) modern water’ and two potential explanatory variables after excluding aquifer systems where the depth to confined conditions was estimated as shallower than 100 m. Significant (Spearman P-value < 0.05) correlations coefficients are in bold

Calculated depth below which most* samples	Average groundwater pumping estimated for
--	---

Formatted: Font: Italic, No underline

Formatted: Font: Italic, No underline

Formatted: Font: Not Bold

Formatted Table

Deleted Cells

Deleted Cells

Deleted Cells

Formatted: Font: Bold

Formatted: Font: Bold

Deleted Cells

Formatted: Font: Bold

Formatted: Font: Bold

Formatted: Font color: Text 1

Formatted: Font: Bold

Formatted: Font: Bold

Formatted: Font color: Text 1

Formatted: Font: 10 pt

contain minimal (<25%) modern water	the year 2015 (mm/year)
most* = 60%	$\rho = 0.34$ (P = 0.0462); sample size of N=34
most* = 70%	$\rho = 0.42$ (P = 0.0148); sample size of N=32
most* = 80%	$\rho = 0.42$ (P = 0.0226); sample size of N=29

* "most" interpreted as 60% (row 2 of table), 70% (row 3 of table) or 80% (row 4 of table)

In similar fashion to our analyses presented in Supplementary Table 2, we completed multiple regression on the rank transforms of our four potential explanatory variables and report the resulting partial regression coefficients (β). We find a similar result in this analysis (where we withheld N=17 aquifer systems identified in Supplementary Table 9) as the result we obtained when we include the N=17 aquifers identified in Supplementary Table 9.

Specifically, the partial regression coefficient associated with annual groundwater withdrawals is significantly (Spearman P value < 0.01) correlated with the depth below which most samples contain minimal (<25%) modern water (Supplementary Table 11). We conclude that our main result—that annual groundwater withdrawals is significantly correlated (defined in this subsection as Spearman P < 0.01) with the depth below which ‘most’ samples contain minimal (<25%) modern water and four potential explanatory variables (where ‘most’ is defined as 60%, 70% or 80%).

Supplementary Note 6. Potential explanations for deep modern groundwater

To better understand the potential mechanisms by which modern groundwater reaches deep depths (while it is still sufficiently young in its age to be considered ‘modern’), we reviewed a number of studies distributed across a subset of our study areas (Supplementary Table 82).

The alternating white/grey shading in the table below delimits generalized categories of features, processes or characteristics (e.g., the first three rows (after the header row) have light grey shading, as all three rows (each row being one study aquifer system) may be receiving artificial recharge via leaking canals or excess irrigation waters—thus, these three rows have the same general category listed in the second column from the left: “Artificial recharge from canals and excess irrigation water may provide additional driving force behind downward-oriented vertical hydraulic gradients, that speed up downward flows of groundwater”). References for quotes in the rightmost column are listed below the table.

Supplementary Table 82. Potential mechanisms by which modern groundwater reaches deep depths while it is sufficiently young in its age to be considered ‘modern’. A single aquifer system (and identical quote) may appear more than once in the table if it is applicable to multiple general processes, features or characteristics of the aquifer system that may potentially play a role in determining vertical variations in modern groundwater (i.e., multiple categories in column two)

Supplementary Table 11. Multiple regression of the rank-transforms coefficients (*p*) describing statistical relationship between “the depth below which most* samples contain minimal (<25%) modern water and four potential explanatory variables after excluding the N=17 aquifers in Supplementary Table 9. Statistically significant (P-value < 0.01) partial regression coefficients (*β*) are in bold

Aquifer system	General process, feature or characteristic of aquifer system that may play a role in determining vertical variations in modern groundwater				Quote from primary literature (underlines added here to highlight components of the quote relevant to the statement in the second column from the left)
Salt Lake Valley	Artificial recharge from canals and excess irrigation water may provide additional driving force behind downward-oriented vertical hydraulic gradients, that speed up downward flows of groundwater				"This age difference is probably affected by the primary recharge area on the west side of the valley being upgradient from two major components of recharge in the area under modern conditions: losses from canals and infiltration from irrigated fields." Thiros et al. (2010)
Utah Lake Valley	Artificial recharge from canals and excess irrigation water may provide additional driving force behind downward-oriented vertical hydraulic gradients, that speed up downward flows of groundwater				"The young component of water in this sample (apparent age of 18 years, fig. 45) combined with an enriched δ18O value relative to δD (fig. 41) indicates that recharge to the WU aquifer in this area is from an evaporated source, such as seepage of unconsumed irrigation water or from canals carrying water from Utah Lake." Cederberg et al. (2008)
Upper Santa Cruz Basin	Artificial recharge from excess irrigation water and urban runoff may provide additional driving force behind downward-oriented vertical hydraulic gradients, that speed up downward flows of groundwater				"Large changes in tritium in areas of focused infiltration indicated enhanced displacement of older groundwaters by modern recharge... Findings from our study also showed anomalously high CFC concentrations in groundwaters, and non-point source pollution from urban runoff or recharge was identified as the most likely source of elevated CFCs." Carlson et al. (2011)
San Joaquin Basin	Calculated depth below which most* samples contain minimal (<25%) modern water	Average precipitation divided by potential evapotranspiration	Average topographic slope (percent)	Average groundwater pumping estimated for the year 2015 (mm/year)	Median thickness of permeable layers
	Artificial recharge from excess irrigation water may provide additional driving			Tritium concentrations somewhat higher than in surrounding areas that are found in the southern San Joaquin Valley are also likely the result of intense pumping and recharge via return flow of imported irrigation water." Visser et al. (2014)	
				"Groundwater classified as young was most	

Formatted: Strong, Font: 10 pt, Not Bold, Font color: Text 1,
 Formatted: Normal
 Formatted: Font: 10 pt,

Deleted Cells
 Inserted Cells
 Formatted: Widow/Orphan control
 Formatted: Widow/Orphan control
 Deleted Cells
 Deleted Cells

Aquifer system	General process, feature or characteristic of aquifer system that may play a role in determining vertical variations in modern groundwater	Quote from primary literature (underlines added here to highlight components of the quote relevant to the statement in the second column from the left)
	force behind downward-oriented vertical hydraulic gradients, that speed up downward flows of groundwater. (unitless) grade)	often located closer to the valley trough than groundwater classified as old (fig. 10C, table 6), possibly because of infiltration of young irrigation water in the San Joaquin Valley..." Bennett et al. (2010)
Albuquerque Basin	Groundwater moves downwards through area(s) of constructed wells, potentially introducing shallower (younger) groundwater into deeper portions of the aquifer near to the well	"Whether enhancement of downward gradients by pumping or by direct movement of water through well bores is the primary mechanism of vertical mixing within the study area, the end result is the presence across broad areas of young, potentially contaminated water at greater depths than would be expected in the absence of deep supply wells..." Bexfield et al. (2012)
San Joaquin Basin	Groundwater moves downwards through area(s) of constructed wells, potentially introducing shallower (younger) groundwater into deeper portions of the aquifer near to the well	"During static conditions, strong vertical hydraulic gradients cause shallow ground water to migrate from the upper part of the well screen to the near bottom of the well screen and into the deep part of the aquifer, where it was stored until the next pumping cycle." Jurgens et al. (2008)
Northern High Plains	Groundwater moves downwards through area(s) of constructed wells, potentially introducing shallower (younger) groundwater into deeper portions of the aquifer near to the well	"Irrigation withdrawals from the confined aquifer result in large downward hydraulic head gradients, creating conditions where water from the unconfined aquifer can move downward to the confined aquifer through boreholes that cross the confining unit" Landon et al. (2010)
Boise Valley and Homedale Murphy Area	Groundwater moves downwards through area(s) of constructed wells, potentially introducing shallower (younger) groundwater into deeper portions of the aquifer near to the well	"Tritium is present in shallow aquifers, such as those underlying the New York Canal (Hutchings and Petrich, 2002b). Tritium is virtually non-existent in deeper, regional ground waters, except where well construction has allowed inter-aquifer mixing" Petrich and Urban (2004)
Houston - Galveston Area	Groundwater moves downwards through area(s) of constructed wells, potentially introducing shallower (younger) groundwater into deeper portions of the aquifer near to the well	"Additionally, there may be nearby wells with multiple screened intervals in the aquifer(s), allowing for leakage of younger water to the older reservoirs of water, providing a conduit for cross-formational flow and biasing the resulting apparent age." Oden and Truini (2013)
San Joaquin Basin	Groundwater moves downwards through area(s) of constructed wells, potentially introducing shallower (younger) groundwater into deeper portions of the aquifer near to the well	"Furthermore, the groundwater age distribution indicated younger water at depth near the industrial supply well (average of 27 years on Fig. 9) relative to the surrounding groundwater by a factor of as much as 100%. These observations indicated that the industrial supply well acted as a vertical conduit for downward flow and nitrate migration and that the impact to groundwater quality extended more than 1,000 ft (300 m) from the well." Gailey (2017)*
Salt Lake Valley	Groundwater flows downward, potentially crossing boundaries between geologic formations with different hydraulic conductivities (i.e., cross formational flow)	"The primary exception is an area of young water (<20 years old) in the middle of the study area cored by wells 23, 33, and 36; ages from these wells are 5 – 15 years younger than ages from wells located immediately to the east, closer to the mountain front. This local reversal of the regional age gradient is consistent with the X_{min} results, which indicate that wells in this area are drawing in locally recharged water from the shallow unconfined aquifer." Manning and Solomon (2005)
Eastern Mississippi Embayment	Groundwater flows downward, potentially crossing boundaries between geologic formations with different hydraulic conductivities (i.e., cross formational flow), in this case, these cross formational flows may be related to discontinuities in a confining unit ('windows' in the aquitard)	"The age-date tracer data provide an indication of the intrinsic vulnerability... vulnerability along these flow paths is related to a combination of factors that include the presence of contaminant sources, aquifer conditions (confined or unconfined), proximity to windows in the confining unit, redox conditions, and the degree to which groundwater withdrawals have induced downward flow from shallow parts of the aquifer system." and "...a component of young water is present in the aquifer at most locations along both flow paths, which is consistent with previous studies at Memphis that documented leakage of shallow water into the Memphis aquifer locally where the overlying confining unit is thin or absent." Kingsbury et al. (2017)

Formatted: Font: 8 pt, Font color: Text 1, English (United States)

Formatted: Font: 8 pt, Font color: Text 1, English (United States)

Aquifer system	General process, feature or characteristic of aquifer system that may play a role in determining vertical variations in modern groundwater	Quote from primary literature (underlines added here to highlight components of the quote relevant to the statement in the second column from the left)
San Pedro Basin	Groundwater flows downward, potentially crossing boundaries between geologic formations with different hydraulic conductivities (i.e., cross formational flow)	"There are a few wells in the lower basin-fill aquifer with anomalous geochemistries that may represent areas of potential mixing or recharge from the floodplain aquifer or upper basin-fill aquifer." Hopkins et al. (2014)
San Luis Valley	Groundwater flows "upward" , potentially crossing boundaries between geologic formations with different hydraulic conductivities (i.e., cross formational flow)	"Because there is upward migration from the upper and lower confined aquifers into the unconfined aquifer in the ancestral sump area, this unconfined aquifer water has a mixed origin." Mayo et al. (2010)
Ocala Uplift	Groundwater flows downward, potentially crossing boundaries between geologic formations with different hydraulic conductivities (i.e., cross formational flow); in this case, these cross formational flows may be related to discontinuities in a confining unit ('windows' in the aquitard)	"The Upper Floridan Aquifer and the surficial aquifer system are separated by a discontinuous clay-rich confining unit. A number of localized surface or buried depressions called sinkholes disrupt this layered geologic framework. Breaches in this clay unit result from localized subsidence activity that occurs when the underlying limestone dissolves, causing the collapse of overlying sediments. Many of these breaches in the intermediate confining unit serve as preferential flow paths to the underlying Upper Floridan Aquifer" and "Depth-dependent and monitoring well sampling results indicate that the highly transmissive zone intersecting the PSW below 49 m bls is hydraulically connected to the surficial aquifer system, probably through sinkholes that breach the confining unit, and results in a mixture of water from the surficial aquifer system and the Upper Floridan Aquifer within the PSW" Landon et al. (2010)
Black Hills Uplift	Permeable conduits formed in carbonate (karst) formations within the aquifer system potentially support relatively fast groundwater transport rates	"Anomalies of young groundwater based on chlorofluorocarbons (CFCs), tritium, and electrical conductivity (EC) indicated fast moving, focused flow and thus the likely presence of conduits" Long et al. (2008). "Tritium was used as a metric to characterize ground-water age in the area near a possible conduit. A straight line fit by linear regression between tritium concentration and distance to the possible conduit shown in figure 1 for the six wells closest to the conduit shows an inverse relation between these two parameters (fig. 15)." Putnam and Long (2007)
Salt Lake Valley	Extensive perforated intervals along wells support mixing of younger (shallower) and older (deeper) groundwaters when water is drawn from the well	"Because public-supply wells generally have long open (screened or perforated) intervals (typically 150-500 ft), the samples likely contain mixtures of water with different ages." Thiros et al. (2010)
Los Angeles Basin	Extensive perforated intervals along wells support mixing of younger (shallower) and older (deeper) groundwaters when water is drawn from the well	"As noted above, the ages reported here are the mean of a mixed age, which may have a broad distribution, especially since the groundwater is in most cases produced from wells with very long screened intervals." Hudson et al. (2002)
Tulare Basin	Extensive perforated intervals along wells support mixing of younger (shallower) and older (deeper) groundwaters when water is drawn from the well	"...wells with long-screens, like many public supply wells, can be screened in both systems and therefore capture varying amounts of pre- and post-1950s groundwater..." Hansen et al. (2018)
most* = 60% Upper Santa Ana Basin	$\beta = -0.12$ ($P = 0.354$) Intentional (i.e., 'managed') aquifer recharge sustains and drives downward-oriented vertical hydraulic gradients, which potentially speed up downward-oriented groundwater flow rates	$\beta = 0.02$ ($P = 0.943$) "In the Hemet area, tritium was detected more frequently in shallower wells (70 percent) than in deeper wells (44 percent; fig. 16A). The more frequent detection of tritium in shallower wells suggests recharge by younger water, primarily from above. However, the shallower wells are also located closer to engineered recharge sources than are the deeper wells ($p = 0.04$; Wilcoxon test)," and "In the comparison of ground-water basins, the relative amount of young water was used as an indicator of engineered recharge." Hamlin et al. (2005)

$\beta = 0.37$ ($P = 0.005$)
 $\beta = 0.49$ ($P = 0.362$)

- Formatted: Widow/Orphan control
- Deleted Cells
- Formatted: Left, Widow/Orphan control
- Formatted Table
- Deleted Cells
- Formatted: Font: 8 pt, Not Bold, Font color: Text 1, English (United States)
- Formatted: Font: 8 pt, Font color: Text 1, English (United States)
- Formatted: Font: 8 pt, Not Bold, Font color: Text 1, English (United States)

Aquifer system	General process, feature or characteristic of aquifer system that may play a role in determining vertical variations in modern groundwater	Quote from primary literature (underlines added here to highlight components of the quote relevant to the statement in the second column from the left)
Los Angeles Basin	Intentional (i.e., 'managed') aquifer recharge sustains and drives downward-oriented vertical hydraulic gradients, which potentially speed up downward-oriented groundwater flow rates	"...intense managed aquifer recharge and deep groundwater pumping draws modern groundwater down to greater depths" Visser et al. (2014)
Salt Lake Valley	Pumping from wells draws recently recharged (i.e., modern) groundwater down to deeper depths	"Withdrawals from the principal aquifer in the area may have allowed recently recharged water to move downward into the aquifer in the vicinity of these wells." Thiros and Manning (2003)
Los Angeles Basin	Pumping from wells draws recently recharged (i.e., modern) groundwater down to deeper depths	"...intense managed aquifer recharge and deep groundwater pumping draws modern groundwater down to greater depths" Visser et al. (2014)
San Joaquin Basin	Pumping from wells draws recently recharged (i.e., modern) groundwater down to deeper depths	"The application of irrigation water and pumping from deep parts of the groundwater system have increased the downward movement of shallow groundwater and stratified both the chemistry and age of water in the aquifer" Jurgens et al. (2016)
Northern High Plains	Pumping from wells draws recently recharged (i.e., modern) groundwater down to deeper depths	"Irrigation withdrawals from the confined aquifer result in large downward hydraulic head gradients, creating conditions where water from the unconfined aquifer can move downward to the confined aquifer through boreholes that cross the confining unit" Landon et al. (2010)
Upper Carbonate Aquifer	Thicker sequences of confining units or relatively low permeability overlying sediment slows downward groundwater flow rates, limiting the depth that modern groundwater reaches	"A comparison of water quality from relatively well protected areas (where the aquifer is overlain by a bedrock confining unit or more than 100 feet of Quaternary-age deposits) and relatively poorly protected areas (where the aquifer is not overlain by a bedrock confining unit or is overlain by less than 100 feet of Quaternary-age deposits) of the Silurian-Devonian and Upper Carbonate aquifers in the study area was performed. Tritium-based ground-water ages were significantly older ($p=0.024$) in relatively well protected areas of the aquifers than in relatively poorly protected areas (fig. 7)" Savoca et al. (1999)
Upper Santa Ana Basin	Unconfined conditions support more widespread modern groundwater, whereas confining units limit groundwater flow rates and thus the ubiquity of modern groundwater	"Tritium was detected more frequently in the unconfined area (97 percent of sampled wells) than in the confined area (59 percent of sampled wells; fig. 22A), indicating that ground water is younger in the unconfined area than in the confined area, and flows laterally from the unconfined to the confined area" Hamlin et al. (2005)
most* = 70% Coachella Valley	$\beta = -0.17$ ($P = 0.205$) Losing rivers serve as a source of modern groundwater recharge, increasing the prevalence of modern groundwater in portions of the aquifer system that are closer to these losing rivers	$\beta = 0.20$ ($P = 0.345$) "Samples collected from the four wells in the Beaumont storage unit had tritium concentrations in excess of 0.2 TU, ranging from 0.5 to 1.9 TU indicating that these wells have received recharge within the past 50 years. Wells 24E2, 28A1, 7E2, and 12K1 are adjacent to stream channels (fig. 36); infiltration along these stream channels probably is the source of the recent recharge to these wells" Lewis et al. (2006)

most* = 80%	$\beta = -0.23$ ($P = 0.078$)	$\beta = 0.25$ ($P = 0.258$)	$\beta = 0.43$ ($P = 0.001$)	$\beta = -0.08$ ($P = 0.709$)
-------------	---------------------------------	--------------------------------	--------------------------------	---------------------------------

* "most" interpreted as 60% (row 2 of table), 70% (row 3 of table) or 80% (row 4 of table)

Bennett, G.L., Fram, M.S., Beltz, K., Jurgens, B.C. (2010). Status and Understanding of Groundwater Quality in the Northern San Joaquin Basin. 2005. California GAMA Priority Basin Project. US Geological Survey Scientific Investigations Report 2010-5175, 96 pp. Accessed May 31, 2022 via <https://pubs.usgs.gov/sir/2010/5175/pdf/sir20105175.pdf>

Bexfield, L.M., Jurgens, B.C., Crilley, D.M., Christenson, S.C. (2012). Hydrogeology, water chemistry, and transport processes in the zone of contribution of a public-supply well in Albuquerque, New Mexico, 2007-9. US Geological Survey Scientific Investigations Report 2011-5182, 114 pp. Accessed July 5, 2022 via <https://pubs.usgs.gov/sir/2011/5182/>

Carlson, M. A., Lohse, K. A., McIntosh, J. C., McLain, J. E. (2011). Impacts of urbanization on groundwater quality and recharge in a semi-arid alluvial basin. Journal of Hydrology, 409, 196-211.

Cederberg, J.R., Gardner, P.M., Thiros, S.A. (2008). Hydrology of Northern Utah Valley, Utah County, Utah, 1975-2005. US Geological Survey Scientific Investigations Report 2008-5197, 128 pp. Accessed May 31, 2022 via <https://pubs.usgs.gov/sir/2008/5197/pdf/sir2008-5197.pdf>

Galley, R.M. (2017). Inactive supply wells as conduits for flow and contaminant migration: conditions of occurrence and suggestions for management. Hydrogeology Journal, 25, 2163-2183.

- Deleted Cells
- Deleted Cells
- Formatted: Widow/Orphan control
- Formatted: Left, Widow/Orphan control
- Formatted Table
- Formatted: Font: 8 pt, Font color: Text 1, English (United States)
- Formatted: Font: 8 pt, Not Bold, Font color: Text 1, English (United States)
- Formatted: Font: 8 pt, Not Bold, Font color: Text 1, English (United States)

- Hamlin, S.N., Belitz, K., Johnson, T. (2005). Occurrence and Distribution of Volatile Organic Compounds and Pesticides in Ground Water in Relation to Hydrogeologic Characteristics and Land Use in the Santa Ana Basin, Southern California. US Geological Survey Scientific Investigations Report 2005-5032, 49 pp. Accessed May 31, 2022 via <https://pubs.usgs.gov/sir/2005/5032/sir2005-5032.pdf>
- Hansen, J.A., Jurgens, B.C., Fram, M.S. (2018). Quantifying anthropogenic contributions to century-scale groundwater salinity changes, San Joaquin Valley, California, USA. *Science of the Total Environment*, 642, 125-136.
- Hopkins, C.B., McIntosh, J.C., Eastoe, C., Dickinson, J.E., Meixner, T. (2014). Evaluation of the importance of clay confining units on groundwater flow in alluvial basins using solute and isotope tracers: the case of Middle San Pedro Basin in southeastern Arizona (USA). *Hydrogeology Journal*, 22, 829-849.
- Hudson, G.B., Moran, J.E., Eaton, G.F. (2002). Interpretation of Tritium-3 Helium Groundwater Ages and Associated Dissolved Noble Gas Results from Public Water Supply Wells in the Los Angeles Physiographic Basin. Report to the California State Water Resources Control Board, 59 pp. Accessed May 31, 2022 via https://water.lnl.gov/sites/water/files/2020-09/cas_lnl_la_orange.pdf
- Jurgens, B.C., Böhlke, J.K., Kauffman, L.J., Belitz, K., Esser, B.K. (2016). A partial exponential lumped parameter model to evaluate groundwater age distributions and nitrate trends in long-screened wells. *Journal of Hydrology*, 543, 109-126.
- Jurgens, B.C., Burow, K.R., Dalgish, B.A., Shelton, J.L. (2008). Hydrogeology, Water chemistry, and factors affecting the transport of contaminants in the zone of contribution of a public-supply well in Modesto, Eastern San Joaquin Valley, California. US Geological Survey Scientific Investigations Report 2008-5156, 94 pp. Accessed May 31, 2022 via <https://pubs.usgs.gov/sir/2008/5156/pdf/sir20085156.pdf>
- Landon, M.K., Jurgens, B.C., Katz, B.G., Eberts, S.M., Burow, K.R., Crandall, C.A. (2010). Depth-dependent sampling to identify short-circuit pathways to public-supply wells in multiple aquifer settings in the United States. *Hydrogeology Journal*, 18, 577-593.
- Kingsbury, J.A., Barlow, J.R., Jurgens, B.C., McMahon, P.B., Carmichael, J.K. (2017). Fraction of young water as an indicator of aquifer vulnerability along two regional flow paths in the Mississippi embayment aquifer system, southeastern USA. *Hydrogeology Journal* 25, 1661-1678.
- Long, A. J., Sawyer, J. F., Putnam, L. D. (2008). Environmental tracers as indicators of karst conduits in groundwater in South Dakota, USA. *Hydrogeology Journal*, 16, 263-280.
- Manning, A. H., Solomon, D. K. (2005). An integrated environmental tracer approach to characterizing groundwater circulation in a mountain block. *Water Resources Research*, 41, W12412.
- Mayo, A. L., Davey, A., Christiansen, D. (2007). Groundwater flow patterns in the San Luis Valley, Colorado, USA revisited: an evaluation of solute and isotopic data. *Hydrogeology Journal*, 15, 383-408.
- Oden, T.D., Truini, M. (2013). Estimated rates of groundwater recharge to the Chicot, Evangeline and Jasper aquifers by using environmental tracers in Montgomery and adjacent counties, Texas, 2008 and 2011. US Geological Survey Scientific Investigations Report 2013-5024, 61 pp. Accessed July 5, 2022 via <https://pubs.usgs.gov/sir/2013/5024/SIR2013-5024.pdf>
- Petrich, C.R., Urban, S.M. (2004). Characterization of ground water flow in the Lower Boise River Basin, Idaho Depart. of Water Resources, 158 pp. Accessed May 31, 2022 via <https://idwr.idaho.gov/wp-content/uploads/sites/2/projects/treasure-valley/TVHP-Characterization.pdf>
- Putnam, L.D., Long, A.J. (2007). Characterization of Ground-Water Flow and Water Quality for the Madison and Minnelusa Aquifers in Northern Lawrence County, South Dakota. US Geological Survey Scientific Investigations Report 2007-5001, 39 pp. Accessed May 31, 2022 via <https://pubs.usgs.gov/sir/2007/5001/pdf/SIR07-5001webSpread.pdf>
- Rewis, D.L., Christensen, A.H., Matti, J.C., Hevesi, J.A., Nishikawa, T., Martin, P. (2006). Geology, Ground-Water Hydrology, Geochemistry, and Ground-Water Simulation of the Beaumont and Banning Storage Units, San Geronio Pass Area, Riverside County, California. US Geological Survey Scientific Investigations Report 2006-5026, 191 pp. Accessed May 31, 2022 via https://pubs.usgs.gov/sir/2006/5026/pdf/sir_2006-5026.pdf
- Savoca, M.E., Sadtler, E.M., Akers, K.K. (1999). Ground-water quality in the eastern part of the Silurian-Devonian and Upper Carbonate Aquifers in the eastern Iowa basins, Iowa and Minnesota, 1996. US Geological Survey Water-Resources Investigations Report 98-4224, 35 pp. Accessed May 18, 2022 via <https://pubs.usgs.gov/wri/1998/wri984224/pdf/wri98-4224.pdf>
- Thiros, S.A., Bexfield, L.M., Anning, D.W., Huntington, J.M., eds. (2010). Conceptual understanding and groundwater quality of selected basin-fill aquifers in the Southwestern United States. US Geological Professional Paper 1781, 288 pp. Accessed May 19, 2022 via <https://pubs.usgs.gov/pp/1781/>
- Thiros, S.A., Manning, A.H. (2003). Quality and sources of ground water used for public supply in Salt Lake Valley, Salt Lake County, Utah, 2001. US Geological Survey Water-Resources Investigations Report 2003-4325, 24 pp. Accessed May 31, 2022 via https://pubs.usgs.gov/pp/1781/pdf/pp1781_section2.pdf
- Visser, A., Moran, J.E., Singleton, M.J., Esser, B.K. (2014). California GAMA Special Study. Geostatistical analysis of groundwater age and other noble gas derived parameters in California groundwater. Lawrence Livermore National Laboratory Report to the State Water Resources Control Board, 44 pp. Accessed May 31, 2022 via https://www.waterboards.ca.gov/rwqcb5/water_issues/owts/2014_10_swrcb_basinwide_gama_rpt.pdf

① Upper Santa Cruz Basin (Tucson Basin): "Findings from our study also showed anomalously high CFC concentrations in groundwaters, and non-point source pollution from urban runoff or recharge was identified as the most likely source of elevated CFCs" quoting Carlson et al. (2011)

② Los Angeles Basin: "...intense managed aquifer recharge and deep groundwater pumping draws modern groundwater down to greater depths" quoting Visser et al. (2014)

③ San Joaquin Basin: "Groundwater classified as young was most often located closer to the valley trough than groundwater classified as old... possibly because of infiltration of young irrigation water in the San Joaquin Valley" quoting Bennett et al. (2010)

④ Utah Lake Valley: "The young component of water in this sample... combined with an enriched $\delta^{18}O$ value relative to δD ... indicates that recharge to the WU aquifer in this area is from an evaporated source, such as seepage of unconsumed irrigation water or from canals carrying water from Utah Lake" quoting Cederberg et al. (2008)

⑤ Tulare Basin: "...wells with long-screens, like many public supply wells, can be screened in both systems and therefore capture varying amounts of pre- and post-1950s groundwater..." quoting Hansen et al. (2018)

⑥ Salt Lake Valley: "Withdrawals from the principal aquifer in the area may have allowed recently recharged water to move downward into the aquifer in the vicinity of these wells" quoting Thiros and Manning (2003)

⑦ Northern High Plains: "Irrigation withdrawals from the confined aquifer result in large downward hydraulic head gradients, creating conditions where water from the unconfined aquifer can move downward to the confined aquifer through boreholes that cross the confining unit" quoting Landon et al. (2010)

⑧ Eastern Mississippi Embayment: "...a component of young water is present in the aquifer at most locations along both flow paths, which is consistent with previous studies at Memphis that documented leakage of shallow water into the Memphis aquifer locally where the overlying confining unit is thin or absent" quoting Kingsbury et al. (2017)

Supplementary Fig. 243. Some of the processes that potentially influence vertical distributions of modern groundwater. Each potential recharge-source/process is labelled (numbers 1-8 on figure). The numbered sources/processes correspond to a quote from a publication (see numbers 1-8 surrounding the schematic; full citations are available on the preceding pages).

Supplementary Note 7. Modern groundwater prevalence in wells defined by the USGS as tapping a confined aquifer

To better understand the potential importance of confining units in determining the prevalence of modern water, we subset our groundwater tritium measurements to examine samples collected from confined aquifers. Specifically, we examined $n=1,831$ groundwater ^3H measurements for water samples collected from a well that the USGS has defined tapping confined conditions (“aqfr_type_cd” defined as “Confined single aquifer” or as “Confined multiple aquifers”). These groundwater ^3H data are presented in panel a below, and their associated modern groundwater fractions are categorized in panel b in the figure below.

Supplementary Fig. 244. Prevalence of modern groundwater in groundwater samples collected from a well defined by the USGS as tapping a confined aquifer. a) Measured tritium (in tritium units) or well waters collected from a well defined as confined by the USGS. High- ^3H water samples collected from confined aquifers are marked by orange and red circles. b) Locations of wells defined as tapping a confined aquifer that have at least 5% modern groundwater (i.e., the *minimum* $F_{\text{Post-1953}}$ exceeds 5% shown as pink

circles in panel b). Aquifers by GebreEgziabher et al. (2022) (<https://www.nature.com/articles/s41467-022-29678-7>) are shown as light grey background polygons, and correspond to the table on the following page.

We grouped the points displayed in panel b in the figure on the previous page according to the aquifer system that they overlie (see aquifer system boundaries basemap in the figure on the previous page); we present the percent of all samples collected from a well tapping a confined aquifer with more than 5% modern water (rightmost column).

Supplementary Table 83. Proportion of samples from confined aquifers with some (more than 5%) modern water

Aquifer system title	Number of well water samples collected from confined aquifer	Percent of all samples collected from a well tapping a confined aquifer that have more than 5% modern water
Dougherty Plain and Marianna Lowlands, Floridan Aquifer System	25	84% of samples from confined aquifers
Springfield Plain, Central Lowland Till Plain	29	83% of samples from confined aquifers
Ocala Uplift, Floridan Aquifer System	63	81% of samples from confined aquifers
Newcastle Till Plain, Central Lowland Till Plain	39	79% of samples from confined aquifers
Western Cambrian-Ordovician Aquifers, Northern Midwest Aquifer System	27	70% of samples from confined aquifers
Volcanic Rift Zone, Eastern Snake River Plain	12	67% of samples from confined aquifers
Balcones Fault Zone, Edwards-Trinity Aquifer System	58	64% of samples from confined aquifers
Los Angeles Basin	27	56% of samples from confined aquifers
Michigan Basin	20	55% of samples from confined aquifers
Salt Lake Valley	81	54% of samples from confined aquifers
Central Wabash and Bloomington Ridged Plain, Central Lowland Till Plain	66	50% of samples from confined aquifers
Mount Vernon Hill County, Central Lowland Till Plain	10	50% of samples from confined aquifers
Black Hills Uplift	53	43% of samples from confined aquifers
Utah Lake Valley	16	38% of samples from confined aquifers
Eastern Silurian-Devonian Aquifers, Northern Midwest Aquifer System	31	35% of samples from confined aquifers
Eastern Carrizo-Wilcox, Carrizo-Wilcox	26	35% of samples from confined aquifers
Northern High Plains, High Plains	38	32% of samples from confined aquifers
Santa Clara-Calleguas Basin	16	31% of samples from confined aquifers
Pearl and Chattahoochee Aquifer System	24	29% of samples from confined aquifers
Black Warrior River Aquifer System	28	29% of samples from confined aquifers
Intermediate Aquifer, Floridan Aquifer System	12	25% of samples from confined aquifers
Southern High Plains, High Plains	21	24% of samples from confined aquifers
Mississippian-Silurian-Devonian Carbonates, Northern Midwest Aquifer System	56	23% of samples from confined aquifers
San Joaquin Basin, California Central Valley	15	20% of samples from confined aquifers
Central Mississippi Embayment, Mississippi Embayment	58	19% of samples from confined aquifers
Vidalia Upland, Floridan Aquifer System	17	18% of samples from confined aquifers
Peedee and Black Creek and Cape Fear Aquifers	23	17% of samples from confined aquifers
Eastern Cambrian-Ordovician Aquifers, Northern Midwest Aquifer System	12	17% of samples from confined aquifers
Bighorn Basin	14	14% of samples from confined aquifers
Eastern Mississippi Embayment, Mississippi Embayment	37	14% of samples from confined aquifers
Delmarva Peninsula, North Atlantic Coastal Plain	24	13% of samples from confined aquifers
Western Carrizo-Wilcox, Carrizo-Wilcox	26	12% of samples from confined aquifers
Tifton Upland, Floridan Aquifer System	64	11% of samples from confined aquifers
Houston-Galveston Area, Gulf Coast Regional Aquifer System	80	10% of samples from confined aquifers
Central Carrizo-Wilcox, Carrizo-Wilcox	11	9% of samples from confined aquifers
Lafayette Area, Gulf Coast Regional Aquifer System	12	8% of samples from confined aquifers
Tulare Basin, California Central Valley	14	7% of samples from confined aquifers
Upper Santa Ana Basin	18	6% of samples from confined aquifers
Tijuana-San Diego	22	5% of samples from confined aquifers
New Jersey Coastal Plain, North Atlantic Coastal Plain	29	3% of samples from confined aquifers
Lower Coastal Plain, Floridan Aquifer System	30	3% of samples from confined aquifers
Ozark Plateaus Aquifer System	30	3% of samples from confined aquifers
Western Mississippi Embayment, Mississippi Embayment	46	2% of samples from confined aquifers
Maryland Western Shores, North Atlantic Coastal Plain	23	0% of samples from confined aquifers
North Carolina and Virginia Coastal Plain, North Atlantic Coastal Plain	22	0% of samples from confined aquifers
Powder River Basin, Northern Great Plains	13	0% of samples from confined aquifers

Supplementary Note 8. Locations of hydrogeologic cross sections

To ensure all of the locally relevant hydrogeologic cross sections depicted within Supplementary Figs. 99-240 intersect with portions of our n=74 study areas, we delineated maps from each publication depicting the location of each cross section. Next, we traced the polyline from these georeferenced maps to develop a geospatial dataset detailing the locations of all of our studied cross sections (see dark blue lines in Supplementary Fig. 245).

Supplementary Fig. 245. Locations of cross sections overlapping each of our 74 study areas. The light blue polygons are the locations of the n=74 study areas we focus on in this project. Each dark blue line represents the location of a cross section depicted in one of the Supplementary Figures in Supplementary Note 3. The label next to each dark blue line states the specific study area for which a given cross section is relevant to. For the original maps that each georeferenced cross section line captures see citations in Supplementary Notes 3.1-3.74.

Formatted: Justified

Formatted: Font: Arial, 11 pt, Not Bold,

Reviewer comments, second round review

Reviewer #1 (Remarks to the Author):

I have reviewed the paper and responses to all review comments. The authors have addressed all my concerns satisfactorily and I have no further comments.

Reviewer #2 (Remarks to the Author):

I think that the authors adequately addressed my original review comments. The revised manuscript is substantially improved. I have two additional comments (below) that the authors may want to consider.

Revisit the use of "pumping-induced drawdown" in the second paragraph and elsewhere. In the groundwater literature, pumping-induced drawdown refers to a decline in hydraulic heads (over time) caused by groundwater extraction. This may impact spatial hydraulic gradients and flow directions. I would avoid using "drawdown" to directly describe the downward migration of modern groundwater.

Regarding the depth to confined conditions, the authors may want to add appropriate caveats to avoid giving the impression that there is a single representative depth. For most of the aquifer systems that were studied, this depth varies considerably across the aquifer area.

REVIEWERS' COMMENTS

Reviewer #1 (Remarks to the Author):

I have reviewed the paper and responses to all review comments. The authors have addressed all my concerns satisfactorily and I have no further comments.

We thank Reviewer #1 for their helpful review of our manuscript.

Reviewer #2 (Remarks to the Author):

I think that the authors adequately addressed my original review comments. The revised manuscript is substantially improved. I have two additional comments (below) that the authors may want to consider.

We thank Reviewer #2 for their helpful review of our manuscript.

Revisit the use of “pumping-induced drawdown” in the second paragraph and elsewhere. In the groundwater literature, pumping-induced drawdown refers to a decline in hydraulic heads (over time) caused by groundwater extraction. This may impact spatial hydraulic gradients and flow directions. I would avoid using “drawdown” to directly describe the downward migration of modern groundwater.

This is an excellent suggestion. We agree that our previous use of “drawdown” could have led to confusion. We have replaced all cases of “drawdown” with “downwelling”, which is used to describe downward groundwater movements to deeper aquifers (e.g., <https://agupubs.onlinelibrary.wiley.com/doi/full/10.1029/2019WR026913>).

Regarding the depth to confined conditions, the authors may want to add appropriate caveats to avoid giving the impression that there is a single representative depth. For most of the aquifer systems that were studied, this depth varies considerably across the aquifer area.

We agree. We added the following sentence to the “Limitations” section of our manuscript:

“Fifth, our estimate of a single value for the depth to confined conditions for a given aquifer system oversimplifies real world heterogeneous conditions. The true depth to confined conditions likely varies, possibly substantially, within each of the n=74 aquifer systems that we study. We stress that the depths to confined conditions presented in Fig. 3b and Supplementary Note 3 are approximations.”

~~Manuscript~~**Revised manuscript** entitled: Modern groundwater reaches deeper depths in heavily pumped aquifer systems

Submitted to: *Nature Communications* on August 10, 2022

Authors: Melissa Thaw ^a, Merhawi GebreEgziabher ^a, Jobel Y. Villafañe-Pagán ^{a,b}, Scott Jasechko ^{a,*}

^a Bren School of Environmental Science and Management, University of California, Santa Barbara, 93106, California, USA

^b Department of Geology, University of Puerto Rico, Mayagüez, 00682, Puerto Rico

Words in abstract: ~~139~~140

Words in main text (~~including~~excluding** figure captions):** ~~3306~~2224

Words in methods: ~~1590~~1652

Number of figures: 3

Number of tables: 0

Number of references: 73

*** Corresponding author:**

Scott Jasechko

University of California at Santa Barbara

2400 Bren Hall, Santa Barbara, California, 93106

Email: jasechko@ucsb.edu

Abstract

Deep groundwater is an important source of drinking water, and can be preferable to shallower groundwaters where they are polluted by surface-borne contaminants. Surface-borne contaminants are disproportionately common in ‘modern’ groundwaters that are made up of precipitation that fell since the ~1950s. Some local-scale studies have suggested that groundwater pumping can draw modern groundwater downward and potentially pollute deep aquifers, but the prevalence of such pumping-induced ~~drawdown~~downwelling at continental scale is not known. Here we analyse thousands of US groundwater tritium measurements to show that modern groundwater tends to reach deeper depths in heavily pumped aquifer systems. These findings imply that groundwater pumping can draw mobile surface-borne pollutants to deeper depths than they would reach in the absence of pumping. We conclude that intensive groundwater pumping can draw ~~shallow~~recently recharged groundwater deeper into aquifer systems, potentially endangering deep groundwater quality.

Main

Introduction

Groundwater resources supply drinking water to billions of people^{1,2}. However, groundwater supplies are vulnerable to pollution from surface-borne contaminants, which can accompany precipitation as it infiltrates the land surface and percolates down to the water table³. Surface-borne contaminants are disproportionately common in groundwater that is made up of relatively recent precipitation^{4,5,6,7,8,9,10} known as ‘modern groundwater’—defined as groundwater comprised of precipitation that fell more recently than the year 1953. Because surface-borne contaminants are disproportionately common in modern groundwater, understanding the processes that control the depth that modern groundwater reaches is important for evaluating water quality risks in shallower versus deeper wells^{11,12}.

Several local- and regional-scale studies have demonstrated that groundwater pumping can speed up downward flow rates and enable groundwater to reach deep depths while it is still young enough to be considered ‘modern’; such pumping-induced ~~drawdown~~downwelling has the potential to also draw shallow pollutants into deep wells used by municipalities and rural communities^{13,14,15}. However, the prevalence of this pumping-induced ~~drawdown~~downwelling of modern groundwater remains poorly understood, as no continent-wide study has tested if modern groundwater reaches deeper depths in places with higher groundwater withdrawal rates.

The objective of this study is to test for statistical relationships between spatial patterns of groundwater withdrawals and spatial patterns of the depth that modern groundwater reaches. To meet our objective, we analyse thousands of US groundwater tritium (³H) measurements (Fig. 1a) to calculate the depth below which modern groundwater is scarce in US aquifer systems (Methods subsection: Groundwater tritium data). Specifically, we calculated the fraction of each well water sample that is comprised of modern groundwater by comparing measured well water ³H activities to historical precipitation ³H time series¹⁶ (Methods subsection: Modern groundwater calculations). Elevated well water ³H activities indicate that modern groundwater is present in a well water sample¹⁷, because most of the precipitation that fell in the US after the year 1953 was artificially enriched in ³H by atmospheric thermonuclear testing^{16,18}. After

Formatted: Font: Not Italic

Formatted: Font: Not Italic

completing our ^3H -based calculations of the proportion of individual well water samples comprised of modern groundwater, we grouped wells located within the same aquifer system (US aquifer geospatial data from ref.¹⁹). These two-dimensional aquifer system areas are underlain by multiple geologic formations of varying hydrogeologic characteristics (for hydrogeologic cross sections for each study area, see Supplementary Figs. 99-240). Next, for each aquifer system, we calculated the depth below which modern groundwater is scarce by calculating the shallowest depth below which most samples (60%, 70% or 80%; Fig. 2) collected from deeper wells contain ‘minimal modern groundwater’ (‘minimal modern groundwater’ defined as well water samples containing less than 25% modern groundwater; Methods subsection: Calculating the depth below which modern groundwaters are scarce). We excluded n=17 aquifer systems where a visual inspection of modern groundwater variations with depth suggests that our approach did not adequately capture the depth below which modern groundwater becomes scarce (Supplementary Note 1; Supplementary Figs. 1-91; Supplementary Table 1). Last, we estimated groundwater withdrawal rates within the boundaries of each study aquifer system by downscaling county-scale groundwater withdrawal data provided by the US Geological Survey²⁰, and tested for spatial correspondence between groundwater withdrawals and the depth below which modern groundwater becomes scarce via rank regression (Methods subsection: Geospatial analyses of potential explanatory variables; see Supplementary Note 2 for further details pertaining to our calculations of annual groundwater withdrawals).

Formatted: Font: Not Italic

Formatted: Font: Not Italic

Results and Discussion

Fig. 1. Well water tritium (^3H) measurements across the contiguous United States. a) Light yellow shades represent lower tritium activities (below 1 tritium unit), orange shades represent mid-range tritium activities (1 to 10 tritium units) and red shades represent well waters with high tritium activities (exceeding 10 tritium units; 1 tritium unit is ~ 0.118 Bq/L). Light grey polygons underlying the yellow-red points are aquifer system boundaries published by GebreEgziabher et al. (ref.⁶⁸; data available via <https://www.hydroshare.org/resource/d2260651b51044d0b5cb2d203d21af08/>). Panels (b-e) display hydrostratigraphy via cross-sections for n=14 of the n=74 aquifer systems that we studied. The presented cross-sections are based on figures and lithologic descriptions presented for the b) Boise Valley and Homedale Area within the broader Snake River Plain in ref.⁶⁴, c) Black Hills Uplift in ref.⁶², d) Williston Basin in ref.⁶³, e) Eastern Dakota Aquifer System in ref.⁶⁴, f) Michigan Basin in ref.⁶⁶, g) Long Island in ref.⁶⁶, h) Castle Hayne Aquifer System in ref.⁶⁷, i) Lower Coastal Plain subarea of the broader Floridan Aquifer System in ref.⁶⁸, j) Carbor-Wellington Aquifer System in ref.⁶⁹, k) Southern High Plains in ref.⁶⁹, l) Mesilla Valley in ref.⁶⁴, m) Santa Clara-Calleguas Basin in ref.⁶², n) Tulare Basin subarea of the broader California Central Valley Aquifer System in ref.⁶³, and the o) Salt Lake Valley in ref.⁶⁴. Cross sections for each of our n=74 study aquifer systems are presented in Supplementary Figs. 99-240 (locations of each of the n=74 hydrogeologic cross-sections we examined are displayed in Supplementary Note 8).

Modern groundwater common at shallow depths

We identified n=74 aquifer systems with sufficient well water ^3H data to quantify the depth below which most (60%, 70% or 80%; Fig. 2a-c) groundwater samples contain minimal modern groundwater. Among our study aquifers, the median depths below which most groundwater samples contain minimal modern groundwater range from 38-75 m (the range of median values derives from our three different quantitative definitions of ‘most’: 60%, 70% or 80%).

The depth below which modern groundwater is scarce is relatively shallow (less than ~ 50 m) in portions of the Gulf Coast Regional Aquifer System (Houston-Galveston and Lafayette sub-areas) and in the Northern Great Plains Aquifer System (Williston Basin and Powder River Basin sub-areas; Fig. 2). By contrast, the depth below which modern groundwater is scarce is relatively deep (greater than ~ 100 m) in Long Island, the Los Angeles Basin, California’s Central Valley (Sacramento, San Joaquin and Tulare sub-areas), and alluvial basins in Arizona (e.g., San Pedro Basin) and California (e.g., Coachella Valley; for individual plots for each aquifer system see Supplementary Figs. 1-91; for map with labelled aquifer system titles see Supplementary Fig. 98).

The observation that modern groundwater tends to be most common at shallow depths has important ramifications for well water quality, as surface-borne contaminants are disproportionately common in modern groundwaters^{4,5,6,7,8,9,10}. Critically, from a water quality risk perspective, we note that most US drinking water wells are perforated at relatively shallow depths where modern groundwater is most common; 55% of domestic water wells are shallower than 50 m, and 84% are shallower than 100 m (ref.²¹); however, we stress that most drinking water wells are domestic water supply wells rather than public water supply wells, and that the latter tend to be deeper than the former. Our finding that most domestic water wells are perforated at relatively shallow depths—where our ^3H data suggest modern groundwaters are most common—implies that the water pumped by many domestic wells is dominated by modern groundwater, which is disproportionately likely to contain surface-borne contaminants.

Drilling domestic water wells to deeper depths may reduce the likelihood of well water contamination events in some areas^{22,23,24,25}. However, drilling deeper wells to avoid shallow contaminated aquifers may

be a stopgap²¹, if pumping from nearby municipal or irrigation wells draws modern groundwater downward and jeopardizes deep groundwater quality. Nevertheless, the prevalence of pumping-induced ~~drawdown~~downwelling at continental scale is not known. Therefore, we calculated correlations between groundwater withdrawals and the depth below which modern groundwater is scarce (see next section entitled: Deeper modern water where groundwater withdrawal rates are high; see Methods subsection; Geospatial analyses of potential explanatory variables).

Fig. 2. Spatial patterns of the depth below which modern groundwater is scarce among US aquifer systems. Yellow-blue shades represent the depth below which 60% (panel a), 70% (panel b) or 80% (panel c) of samples contain minimal modern groundwater (defined here as well water samples containing less than 25% modern groundwater). **a)** The median depth below which more than 60% of samples contain minimal modern groundwater among n=74 studied aquifer systems (with sufficient groundwater-³H data) is 38 meters, and the lower-upper quartile range is 12-92 meters. **b)** The median depth below which more than 70% of samples contain minimal modern groundwater among n=68 studied aquifer systems is 60 meters, and the lower-upper quartile range is 25-113 meters. **c)** The median depth below which more than 80% of samples contain minimal modern groundwater among n=61 studied aquifer systems is 75 meters, and the lower-upper quartile range is 49-149 meters. For a map with labels identifying aquifer systems see Supplementary Fig. 98.

Deeper modern water where groundwater withdrawal rates are high

We find a significant (Spearman P-value < 0.01) positive correlation between annual groundwater withdrawals and the depth below which modern groundwater is scarce (Fig. 3a). Spearman rank correlation coefficients (ρ) range from $\rho = 0.39$ to $\rho = 0.42$ (all statistically significant at P-value < 0.01; the range of ρ values derives from three correlation coefficients, each based on the depth below which 60%, 70% or 80% of samples contain less than 25% modern groundwater; Supplementary Table 78). Our finding suggests that modern groundwater tends to reach deeper depths in aquifer systems that are heavily pumped.

It is plausible that shallow low-permeability geologic formations could limit the depth below which modern groundwater is scarce and influence the effects of pumping on vertical variability in groundwater age²⁶. We estimated the depth to confined conditions for each of our n=74 study areas by analysing (i) vertical variations in the proportion of wells defined as tapping confined aquifers by the US Geological Survey, (ii) hydrogeologic cross sections derived from local-scale studies, and (iii) statements pertaining to confined conditions from local-scale studies (see Fig. 3b; for detailed approach for estimating depth to confined conditions for each of our n=74 study aquifers see Supplementary Notes 3.1 to 3.74). We show that the depth below which modern groundwater is scarce tends to be deeper in aquifer systems

Formatted: Font: Not Italic

Formatted: Font: 10 pt

Formatted: Space Before: 0 pt

characterized by thicker unconfined zones (i.e., aquifers with a relatively high (i.e., deep) depth to confined conditions; see Supplementary Note 4).

Therefore, this significant positive correlation between annual groundwater withdrawals and the depth below which modern groundwater is scarce (Fig. 3a) could arise spuriously if groundwater pumping tends to be higher in aquifer systems characterized by thick unconfined zones. To account for potential interrelationships between groundwater withdrawals and the depth to confined conditions, we completed multiple regression of the rank transforms of each explanatory variable²⁷ (Supplementary Notes 4 and 5). The resulting partial regression coefficients (β)_—that describe the statistical relationship between groundwater withdrawals and the depth below which modern groundwater is scarce_—remain positive and significant (β values range 0.29 to 0.34; all significant at Spearman P-value < 0.05; see Supplementary Table 79). This analysis suggests that our finding that modern groundwater tends to reach deeper depths in aquifer systems that are heavily pumped holds even after accounting for differences in the depth to confined conditions among our study aquifers.

Our pan-US statistical analyses are consistent with local-scale research in California’s Central Valley^{28,29}, where groundwater pumping **drawdowns** young and shallow groundwater deeper into the aquifer system; our results suggest that pumping-induced **drawdown/downwelling** is not unique to California’s Central Valley and is likely occurring in other heavily pumped US aquifer systems. Though we find that modern groundwater tends to reach deeper depths in heavily pumped aquifer systems (Fig. 3a), we emphasize the moderate strength of the correlation and the high proportion of unexplained variance (see substantial scatter in points in Fig. 3a). We also emphasize that the groundwater withdrawal data that we analyse²¹ are highly uncertain; further, the county-scale at which these data are available limit their local relevance and complicate geospatial analyses (see Methods subsection entitled: *Limitations*). There is considerable room for improvement in US groundwater withdrawal data³⁰. Should better groundwater withdrawal data become available, these data would enable a better assessment of the statistical relationship between groundwater withdrawals and the depths that modern groundwaters reach.

Fig. 3. The depth below which modern groundwater is scarce tends to be deeper where annual groundwater withdrawals are high. a) Relationship between estimated annual groundwater withdrawals²¹

and the depth below which modern groundwater is scarce. Each point represents one aquifer system. Annual groundwater withdrawals were normalized by aquifer area, such that the units (mm/year) can be interpreted as all withdrawn groundwater during the year 2015 expressed as saturated layer if it were spread evenly across the study area (for study area boundaries see panel b). The Spearman rank correlation coefficient (ρ) of the data presented in panel a is $\rho = 0.42$ (Spearman P-value < 0.001). The y-axis values corresponding to each coloured point are the depth below which $>70\%$ of samples have minimal (less than 25%) modern groundwater; vertical error bars extend to the depth below which $>60\%$ of samples have minimal (less than 25%) modern groundwater (shallower depth—i.e., top of grey error bar as displayed in plot) and the depth below which $>80\%$ of samples have minimal (less than 25%) modern groundwater (deeper depth—i.e., bottom of grey error bar as displayed in plot; see legend in upper left corner of plot). We only plot the $n=74$ points representing the aquifer systems for which we had sufficient data to determine the depth below which $>70\%$ of samples have minimal (less than 25%) modern groundwater. Points are colour-coded by the estimated depth to confined conditions (see legend in panel b). The significant correlation between the depth below which modern groundwater becomes scarce and annual groundwater withdrawals remains significant when we consider potential statistical inter-relationships between groundwater withdrawals and the depth to confined conditions (Supplementary Note 4) or when we exclude aquifer systems with shallow depths to confined conditions (Supplementary Note 5). **b)** Map of our estimated depth to confined conditions in each of our $n=74$ study aquifer systems. Each polygon on the map represents one study area. Light blue colors represent shallower depths to confined conditions; darker blue shades represent deeper depths to confined conditions. Depths to confined conditions were estimated on the basis of up to three data sources: (i) USGS defined well conditions (i.e., wells defined as tapping unconfined versus confined conditions by the US Geological Survey), (ii) digitization and evaluation of hydrogeologic cross sections derived from local scale reports, and (iii) quotes within local scale reports pertaining to the prevalence of confined conditions. For details on our estimated depths to confined conditions for each of our $n=74$ study areas, see Supplementary Note 3. For a comparison of the depth to confined conditions for each of our study areas see Supplementary Fig. 242.

Potential explanations for deep modern groundwater

While our tritium dataset cannot identify the specific mechanisms that transmit modern groundwater to deeper depths where groundwater withdrawal rates are high, there are a number of potential mechanisms that may help explain our main finding—that modern groundwater tends to reach deeper depths in aquifer systems that are heavily pumped (Fig. 4; Supplementary Note 6).

First, pumping from deep aquifers can alter vertical hydraulic gradients and speed up downward flows of shallow groundwaters^{13,31,32,33,34}; the existence of preferential flow pathways through permeable geologic structures (e.g., discontinuities in aquitards at faults or permeable intercalations³⁵) may further enhance pumping-induced downward transport of modern groundwater³⁶. Second, the magnitude of downward-oriented vertical hydraulic gradients—which drive modern groundwater to deep depths—may be increased by artificial groundwater recharge associated with land use and water management²⁸ (e.g., infiltration of excess irrigation waters^{37,38}, leakage from urban water infrastructure³⁹ or canals⁴⁰). Spreading basins and intentional flooding have been applied via ‘managed aquifer recharge’ projects for decades in several heavily pumped aquifer systems in California and Arizona^{41,42}, potentially increasing downward groundwater flow rates in these areas. Recharge induced by such processes can sustain high-magnitude and downward-oriented vertical hydraulic gradients and may drive modern groundwaters to deeper depths. Third, modern groundwaters may be present in some deep aquifers because they were intentionally injected via aquifer storage and recovery projects, but these projects are not common in many of our study areas⁴³; this mechanism is speculative but plausible. Fourth, modern groundwater may

be transmitted relatively rapidly to deep depths in heavily pumped aquifer systems that also host poorly sealed wells, which can create conduits that enable rapid movement of shallow modern groundwater to deep aquifers^{35,44,45,46}.

To better understand the potential for confining units to limit the depth that modern groundwater reaches, we examined the prevalence of modern groundwater in n=1,831 groundwater samples where the well has been defined by the US Geological Survey as tapping a confined aquifer (well metadata field `aqfr_type_cd` specifies 'Confined single aquifer' or 'Confined multiple aquifers'). We find that about one-third of groundwater samples collected from a well that is perforated in a confined aquifer (n=592) contain more than 5% modern water, highlighting that modern water can enter wells screened in confined aquifers (Supplementary Note 7; Supplementary Fig. 244).

Differences among our study aquifers in pumping well depths, well integrity, land use activities and managed aquifer recharge practices likely contribute the observed variability in the depth below which modern groundwater becomes scarce in our study areas. Identifying locally relevant mechanisms that may rapidly transmit shallow groundwater to deep depths is important to create strategies to protect deep groundwater quality.

Formatted: Font: Bold

Fig. 4. Schematic of some of the processes that may influence vertical distributions of modern groundwater. Artificial groundwater recharge (left side of schematic) can increase the magnitude of downward-oriented vertical hydraulic gradients and potentially drive modern groundwater to deeper depths. Artificial recharge can derive from urban waters (e.g., Tucson Basin⁶⁶), spreading basins associated with managed aquifer recharge projects (e.g., Upper Santa Ana Basin⁶⁶), excess irrigation waters (e.g., San

Joaquin Basin⁶⁷), or leaky surface water conveyance infrastructure such as canals (e.g., Utah Lake Valley⁶⁸). The construction of groundwater wells and their use via pumping (right side of schematic) may help modern waters enter deeper wells. For example, deep wells that have long perforated intervals may draw both shallow (disproportionately modern) and deep (disproportionately pre-modern) groundwater (e.g., Tulare Basin⁶⁹). Further, pumping from wells can alter vertical hydraulic gradients, potentially drawing shallow modern groundwater to deeper depths (e.g., Salt Lake Valley⁷⁰). The modern groundwater may move downward to deeper depths in the aquifer system via natural pore spaces (e.g., discontinuities in aquitards; e.g., Eastern Mississippi Embayment³⁶) or via conduits created by constructed wells themselves (e.g., Northern High Plains⁷⁴). For further discussion and schematics of modern groundwater distributions see refs.^{60,72,73}. For a review of some of the studies that have posited one or more of these mechanisms as potential explanations for the distribution of modern groundwater see Supplementary Note 6. For a more expansive version of this figure see Supplementary Fig. 243.

Pumping-induced ~~drawdown~~downwelling may impact groundwater quality

Although our tritium data cannot identify specific transport mechanisms for each of the dozens of aquifer systems we studied, our analyses demonstrate a moderately strong statistical relationship that suggests pumping may lead modern groundwaters to reach deeper depths than they would flow to naturally (Fig. 3a).

Where modern groundwater is contaminated, pumping-induced ~~drawdown~~downwelling of these groundwaters can threaten deep groundwater quality; however, we stress that many contaminants flow at considerably slower rates than the groundwater itself due to, for example, retardation via adsorption. The ~~drawdown~~downwelling of modern groundwater may also have indirect impacts on groundwater quality. For example, in the Red River Delta (Vietnam), intensive pumping of deep aquifers has likely increased aqueous arsenic concentrations both directly via the ~~drawdown~~downwelling of shallow arsenic-rich groundwaters to deeper depths³², and also indirectly as these downward-flowing groundwaters contribute arsenic-mobilizing solutes⁴⁷. Because deep aquifers tend to require millennia to flush, contamination of deep groundwaters can be especially challenging to remedy, and some groundwater remediation technologies are not well suited for deep groundwater.

Deep groundwater is a globally important water supply, and its value is expected to grow where shallower groundwater stores and qualities are declining and deteriorating. We demonstrate that modern groundwater tends to reach deeper depths in heavily pumped aquifer systems, signalling that pumping can rearrange groundwater flowpaths and impact deep groundwater quality.

Methods

Groundwater tritium data

We downloaded US well water tritium measurements from the water quality portal (<https://www.waterqualitydata.us/portal> accessed April 20, 2021). We excluded tritium measurements that were below analytical detection limits if the reported detection limit exceeded 0.8 tritium units (where 1 tritium unit equals 0.118 Bq/L). We excluded all measurements of media that were not categorized as ‘Groundwater’ (field code: ‘ActivityMediaSubdivisionName’). We deleted one record with a code reading ‘Systematic Contamination’. We converted negative values to zeros. We excluded one well water measurement with an unlikely ‘ActivityStartDate’ value of September 28, 1900. If more than one tritium measurement was available for a single well, we analysed only the most recent record (i.e., the

Formatted: Font: 16 pt

measurement with the most recent date) and excluded the other measurements, to ensure each well was only counted once in our analysis.

We excluded all records that did not report a numeric well depth value (i.e., both of the following fields were blank: ‘WellDepthMeasure/MeasureValue’ and ‘WellHoleDepthMeasure/MeasureValue’) or recorded a value of zero for the well depth. If a non-zero ‘WellDepthMeasure/MeasureValue’ was available, we prioritized analysis of this well depth value; otherwise, we included the value of ‘WellHoleDepthMeasure/MeasureValue’ as the well depth. We emphasize that both values (i.e., ‘WellDepthMeasure/MeasureValue’ and ‘WellHoleDepthMeasure/MeasureValue’) are likely to be approximations because of the inherent uncertainties associated with well construction. We also highlight that our dataset does not provide information about perforated intervals for all of our wells, requiring us to analyse total well depths rather than screen depth intervals and introducing uncertainty into our analyses (see (ii) in the Limitations section below).

Modern groundwater calculations

We downloaded geospatial time series of estimated precipitation tritium for the contiguous US (from <https://www.sciencebase.gov/catalog/item/5af49307e4b0da30c1b44e10>; see ref.¹⁶). For each groundwater well for which we have tritium data, we identified the nearest precipitation tritium time series grid (in most cases, the well location lies within a precipitation tritium time series grid cell). Next, we decay-corrected all values in the precipitation tritium time series to determine the range of possible ‘net present’ tritium activities relevant to a given groundwater tritium measurement (following method by ref.⁴⁸). To account for the strong likelihood that at least some hydrodynamic dispersion takes place as groundwater flows within aquifer systems, we smoothed these decay-corrected precipitation tritium time series by a five-year running average (following methods by ref.⁴⁹). We then calculated the maximum and minimum decay-corrected post-1953 precipitation tritium values from these running averages; hereafter these values are used as high and low values for the term ${}^3H_{post-1953}$. We then straightforwardly calculated the fraction of each groundwater sample comprised of ‘modern groundwater’ derived from precipitation that fell since the year 1953 ($F_{post-1953}$) following ref.⁴⁸:

$$F_{post-1953} = \frac{{}^3H_{sample} - {}^3H_{pre-1953}}{{}^3H_{post-1953} - {}^3H_{pre-1953}} \quad (1)$$

${}^3H_{sample}$ represents the measured tritium activity in the well water. ${}^3H_{post-1953}$ and ${}^3H_{pre-1953}$ represent the estimated local precipitation tritium activities¹⁶ after correcting for radioactive decay to the time of sampling. ${}^3H_{post-1953}$ represents the decay-corrected precipitation tritium for years between 1953 and the date the sample was analysed (applying the aforementioned five-year running average). ${}^3H_{pre-1953}$ represents the decay-corrected tritium activity for precipitation that fell prior to 1953 (assumed to be zero, as most of groundwater 3H samples in our dataset were analysed after multiple tritium-half-lives have elapsed since 1953, and pre-1953 precipitation tritium activities in the US were likely less than ~10 tritium units⁵⁰).

Calculating the depth below which modern groundwaters are scarce

We then created a binary categorization for each sample, defining each sample as either (i) containing minimal modern groundwater, defined as samples with a maximum $F_{post-1953}$ value of less than 0.25 (i.e., less than 25% of the sample is comprised of modern water), or (ii) samples with a maximum $F_{post-1953}$

value exceeding 0.25. The maximum $F_{post-1953}$ values were determined for each sample using the minimum ${}^3H_{post-1953}$ values in equation 1 (i.e., entering the minimum ${}^3H_{post-1953}$ value into equation 1 yields the maximum $F_{post-1953}$ value for a given groundwater sample).

First, we grouped all groundwater tritium samples by the aquifer system that the well lies within. To avoid analysing the most data-sparse aquifers, we only analysed aquifer systems with (a) at least $n=10$ tritium measurements, and (b) at least a 100 m vertical offset between the total depths of the shallowest well and the deepest well for which we have groundwater tritium measurements.

Second, for each well within each study aquifer, we calculated the proportion of all samples that derive from wells that are deeper than a given depth that contain minimal modern water (i.e., categorization (i) above). We did not calculate these proportions if fewer than five wells were deeper than the depth of interest (i.e., we require a minimum of five data points to calculate a value of 'the proportion of wells with deeper depths that contain minimal (i.e., maximum $F_{post-1953}$ is less than 25% modern water)'). For each aquifer system, we determined the depth at which 60%, 70%, or 80% of samples derived from wells deeper than this depth contain less than 25% modern water (these depths presented in Figs. 1 and 2).

Third, to further scrutinize our results, we visually inspected individual plots of modern groundwater prevalence with depth for each of our study aquifers (Supplementary Figs. 1-91). We identified $n=17$ aquifer systems where our approach (i.e., calculating the depth below which 60%, 70% or 80% of samples contain less than 25% modern groundwater) provided an imperfect estimate of the depth below which modern groundwater becomes scarce due (a) a lack of groundwater tritium data at depth, or (b) a lack of a consistent decline in modern groundwater fractions with depth (Supplementary Table 1 and Supplementary Figs. 4, 6, 9, 12, 16, 19, 33, 37, 44, 52, 54, 59, 71, 79, 82, 87, 88). We excluded these $n=17$ aquifer systems from our analyses.

Geospatial analyses of potential explanatory variables

For each study aquifer system with sufficient groundwater tritium data for analyses, we compared our estimates of 'the depth below which modern groundwater is scarce' (i.e., values in Fig. 2) to two potential explanatory variables: (a) annual groundwater withdrawals (from: <https://www.sciencebase.gov/catalog/item/get/5af3311be4b0da30c1b245d8>), and (b) the depth to confined conditions.

(a) To estimate groundwater withdrawals within each of our study aquifers (i.e., potential explanatory variable (a)), we weighted our county-scale groundwater withdrawal data by the proportion of the aquifer system's area covered by an individual county area (i.e., providing greater statistical weight to counties that cover more area within the aquifer system boundaries. For further details see Supplementary Note 2.

(b) To estimate the depth to confined conditions within each of our study aquifers (i.e., potential explanatory variable (b)), we analysed (i) vertical variations in the proportion of wells defined as tapping confined aquifers by the USGS (depth to confined conditions estimated as the shallowest depth where both (a) the fraction of wells that tap confined conditions exceeds 80%, and (b) more than 80% of wells at deeper depths are classified as tapping confined conditions), (ii) hydrogeologic cross sections derived from local-scale studies, and (iii) statements pertaining to confined conditions from local-scale studies. For further details see Supplementary Note 3.

We then calculated rank correlation coefficients describing variability in our calculated values for the depth below which modern groundwater is scarce and these two explanatory variables (see Supplementary Note 4). Our calculations include independent correlations for each potential explanatory variable (Supplementary Table 78), and also a multiple regression of the rank transforms to account for potential interrelationships among the explanatory variables themselves²⁷ (Supplementary Table 79).

Limitations

Our analyses have a number of limitations, four of which are described here.

(i) First, the groundwater withdrawal data²⁰ we analysed are highly uncertain and available only at county-scale. The substantial uncertainties in groundwater withdrawal data arise in part due to the lack of widespread groundwater withdrawal metering data, which limits our confidence in these withdrawal estimates (i.e., x-axis values in Fig. 3a). Further, the county-scale at which these groundwater withdrawal data are reported²⁰ complicates our interpretation of groundwater withdrawal data and required us to downscale groundwater withdrawal data to finer spatial resolution, introducing uncertainty in the process. We emphasize that the groundwater withdrawal estimates reported in Fig. 3a and our correlation coefficients are uncertain, and that the correlation coefficients we present would differ were better groundwater withdrawal data to become available.

(ii) Second, our evaluation of the depths below which modern groundwater is scarce is limited by the lack of perforated interval data for some of the wells where groundwater tritium measurements have been made. Our calculations are thus necessarily limited to total well depth data, which are, if anything, an overestimate of the depth of the groundwater sampled by the well. Wells that are deeper than the total depths that modern groundwater penetrates may pump modern groundwater because some of these wells will be perforated at shallower depths where modern groundwater is more abundant, potentially leading to overestimations of the depth below which modern groundwaters penetrate in some well with extensive perforated intervals.

(iii) Third, we cannot evaluate the depths at which groundwater withdrawals take place, as the data we analyse²⁰ are two-dimensional data products that do not contain information about the depths at which pumping rates are higher versus lower. However, all of our study aquifers have at least some recorded wells²¹ with depths that exceed the depth below which modern groundwater is scarce, implying a potential for groundwater withdrawals to play a role in determining the depth at which modern groundwater becomes scarce.

(iv) Fourth, we analyse groundwater withdrawals for the year 2015 but acknowledge that groundwater pumping that took place prior to the year 2015 has also likely contributed to the depth distribution of modern groundwater.

(v) Fifth, our estimate of a single value for the depth to confined conditions for a given aquifer system oversimplifies real world heterogeneous conditions. The true depth to confined conditions likely varies, possibly substantially, within each of the n=74 aquifer systems that we study. We stress that the depths to confined conditions presented in Fig. 3b and Supplementary Note 3 are approximations.

Data Availability. Aquifer system boundaries from ref.¹⁹ are available to download via CUAHSI HydroShare: <https://www.hydroshare.org/resource/d2260651b51044d0b5cb2d293d21af08/>. Groundwater withdrawal estimates (county-scale) are available via <https://pubs.er.usgs.gov/publication/cir1441>. Well water tritium data are available via <https://www.waterqualitydata.us/portal>.

Code availability. The analyses presented here do not depend on specific code. Our approach can be reproduced following the procedures described in the Methods.

References

1. Shah, T. Groundwater and human development: challenges and opportunities in livelihoods and environment. *Water Science and Technology* **51**, 27–37 (2005).
2. Margat, J. & van der Gun, J. Groundwater around the world: A geographic synopsis. CRC Press (2013).
3. Foster, S.S.D. & Chilton, P.J. Groundwater: the processes and global significance of aquifer degradation. *Philosophical Transactions of the Royal Society of London. Series B: Biological Sciences* **358**, 1957–1972 (2003).
4. Baran, N., Mouvet, C. & Négrel, P. Hydrodynamic and geochemical constraints on pesticide concentrations in the groundwater of an agricultural catchment (Brévilles, France). *Environmental Pollution* **148**, 729–738 (2007).
5. Brown, K. B., McIntosh, J. C., Rademacher, L. K. & Lohse, K. A. Impacts of agricultural irrigation recharge on groundwater quality in a basalt aquifer system (Washington, USA): a multi-tracer approach. *Hydrogeology Journal* **19**, 1039–1051 (2011).
6. Qin, D., Qian, Y., Han, L., Wang, Z., Li, C. & Zhao, Z. Assessing impact of irrigation water on groundwater recharge and quality in arid environment using CFCs, tritium and stable isotopes, in the Zhangye Basin, Northwest China. *Journal of Hydrology* **405**, 194–208 (2011).
7. Pang, Z., Yuan, L., Huang, T., Kong, Y., Liu, J. & Li, Y. Impacts of human activities on the occurrence of groundwater nitrate in an alluvial plain: a multiple isotopic tracers approach. *Journal of Earth Science* **24**, 111–124 (2013).
8. Han, D., Cao, G., McCallum, J. & Song, X. (2015). Residence times of groundwater and nitrate transport in coastal aquifer systems: Daweijia area, northeastern China. *Science of the Total Environment* **538**, 539–554 (2015).
9. Visser, A., Moran, J. E., Hillemonds, D., Singleton, M.J., Kulongoski, J.T., Belitz, K. & Esser, B.K. Geostatistical analysis of tritium, groundwater age and other noble gas derived parameters in California. *Water Research* **91**, 314–330 (2016).
10. El Mountassir, O., Bahir, M., Ouazar, D., Chehbouni, A. & Carreira, P.M. Temporal and spatial assessment of groundwater contamination with nitrate using nitrate pollution index (NPI), groundwater pollution index (GPI), and GIS (case study: Essaouira basin, Morocco). *Environmental Science and Pollution Research* 1–18 (2021).
11. Zhang, X., Xu, Z., Sun, X., Dong, W. & Ballantine, D. Nitrate in shallow groundwater in typical agricultural and forest ecosystems in China, 2004–2010. *Journal of Environmental Sciences* **25**, 1007–1014 (2013).
12. Egbueri, J. C. & Mgbenu, C. N. Chemometric analysis for pollution source identification and human health risk assessment of water resources in Ojoto Province, southeast Nigeria. *Applied Water Science* **10**, 1–18 (2020).

13. Kagabu, M., Shimada, J., Delinom, R., Nakamura, T. & Taniguchi, M. Groundwater age rejuvenation caused by excessive urban pumping in Jakarta area, Indonesia. *Hydrological Processes* **27**, 2591–2604 (2013).
14. Han, D.M., Song, X.F., Currell, M.J., Yang, J.L. & Xiao, G.Q. Chemical and isotopic constraints on evolution of groundwater salinization in the coastal plain aquifer of Laizhou Bay, China. *Journal of Hydrology* **508**, 12–27 (2014).
15. Taufiq, A., Hosono, T., Ide, K., Kagabu, M., Iskandar, I., Effendi, A.J. & Hutasoit, L.M. Shimada, J. Impact of excessive groundwater pumping on rejuvenation processes in the Bandung basin (Indonesia) as determined by hydrogeochemistry and modeling. *Hydrogeology Journal* **26**, 1263–1279 (2018).
16. Michel, R.L., Jurgens, B.C. & Young, M.B. Tritium deposition in precipitation in the United States, 1953–2012. US Geological Survey Scientific Investigations Report 2018-5086, 19 pp. <https://pubs.er.usgs.gov/publication/sir20185086> (USGS, 2018).
17. Gleeson, T., Befus, K.M., Jasechko, S., Luijendijk, E. & Cardenas, M.B. The global volume and distribution of modern groundwater. *Nature Geoscience* **9**, 161–167 (2016).
18. Doney, S.C., Glover, D.M. & Jenkins, W.J.A model function of the global bomb tritium distribution in precipitation, 1960–1986. *Journal of Geophysical Research: Oceans* **97**, 5481–5492 (1992).
19. GebreEgziabher, M., Jasechko, S. & Perrone, D. Widespread and increased drilling of wells into fossil aquifers in the USA. *Nature Communications* **13**, 2129 <https://doi.org/10.1038/s41467-022-29678-7> (2022).
20. Dieter, C.A. et al. Estimated use of water in the united states in 2015. US Geological Survey Circular 1441, <https://doi.org/10.3133/cir1441> (USGS, 2018).
21. Perrone, D. & Jasechko, S. Deeper well drilling an unsustainable stopgap to groundwater depletion. *Nature Sustainability* **2**, 773–782 (2019).
22. Clark, B.R., Hart, R.M. & Gurdak, J.J. Groundwater availability of the Mississippi Embayment. US Geological Survey Professional Paper 1785, 62 pp. <https://pubs.usgs.gov/pp/1785/> (USGS, 2011).
23. Zhang, X., Xu, Z., Sun, X., Dong, W. & Ballantine, D. Nitrate in shallow groundwater in typical agricultural and forest ecosystems in China, 2004–2010. *Journal of Environmental Sciences* **25**, 1007–1014 (2013).
24. Egbueri, J. C. & Mgbenu, C. N. Chemometric analysis for pollution source identification and human health risk assessment of water resources in Ojoto Province, southeast Nigeria. *Applied Water Science* **10**, 1–18 (2020).
25. Wirmvem, M.J., Ohba, T., Nche, L.A., Kamtchueng, B.T., Kongnso, W.E., Mimba, M.E., Bafon, T.G., Yaguchi, M., Takem, G.E., Fantong, W.Y. & Ako, A.A. Effect of diffuse recharge and wastewater on groundwater contamination in Douala, Cameroon. *Environmental Earth Sciences* **76**, 1–23 (2017).
26. Zinn, B. A., & Konikow, L.F. Potential effects of regional pumpage on groundwater age distribution. *Water Resources Research* **43**, W06418 (2007).
27. Iman, R. L. & Conover, W. J. The use of the rank transform in regression. *Technometrics* **21**, 499–509 (1979).
28. Jagucki, M.L., Jurgens, B.C., Burow, K.R. & Eberts, S.M. Assessing the vulnerability of public-supply wells to contamination—Central Valley aquifer system near Modesto, California: US

- Geological Survey Fact Sheet 2009–3036, 6 pp. <https://pubs.usgs.gov/fs/2009/3036/> (USGS, 2009).
29. Levy, Z.F., Jurgens, B.C., Burow, K.R., Voss, S.A., Faulkner, K.E., Arroyo-Lopez, J.A. & Fram, M.S. Critical aquifer overdraft accelerates degradation of groundwater quality in California's Central Valley during drought. *Geophysical Research Letters* **48**, e2021GL094398 (2021).
 30. Perrone, D., Hornberger, G., van Vliet, O. & van der Velde, M. A review of the United States' past and projected water use. *JAWRA Journal of the American Water Resources Association* **51**, 1183–1191 (2015).
 31. Lapworth, D.J., Das, P., Shaw, A., Mukherjee, A., Civil, W., Petersen, J.O., Goody, D.C., Wakefield, O., Finlayson, A., Krishan, G. & Sengupta, P. Deep urban groundwater vulnerability in India revealed through the use of emerging organic contaminants and residence time tracers. *Environmental Pollution* **240**, 938–949 (2018).
 32. Winkel, L. H., Trang, P.T.K., Lan, V.M., Stengel, C., Amini, M., Ha, N.T., Viet, P.H., Berg, M. Arsenic pollution of groundwater in Vietnam exacerbated by deep aquifer exploitation for more than a century. *Proceedings of the National Academy of Sciences* **108**, 1246–1251 (2011).
 33. Khan, M.R., Koneshloo, M., Knappett, P.S., Ahmed, K.M., Bostick, B.C., Mailloux, B.J., Mozumder, R.H., Zahid, A., Harvey, C.F., Van Geen, A. & Michael, H.A. Megacity pumping and preferential flow threaten groundwater quality. *Nature Communications* **7**, 12833 (2016).
 34. Cognac, K. E. & Ronayne, M. J. Changes to inter-aquifer exchange resulting from long-term pumping: implications for bedrock groundwater recharge. *Hydrogeology Journal* **28**, 1359–1370 (2020).
 35. Santi, P.M., McCray, J.E. & Martens, J.L. Investigating cross-contamination of aquifers. *Hydrogeology Journal* **14**, 51–68 (2006).
 36. Kingsbury, J.A., Barlow, J.R., Jurgens, B.C., McMahon, P.B. & Carmichael, J.K. Fraction of young water as an indicator of aquifer vulnerability along two regional flow paths in the Mississippi embayment aquifer system, southeastern USA. *Hydrogeology Journal* **25**, 1661–1678 (2017).
 37. Scanlon, B.R., Reedy, R.C., Stonestrom, D.A., Prudic, D.E. & Dennehy, K.F. Impact of land use and land cover change on groundwater recharge and quality in the southwestern US. *Global Change Biology* **11**, 1577–1593 (2005).
 38. Ochoa, C.G., Fernald, A.G., Guldan, S.J., Tidwell, V.C. & Shukla, M.K. Shallow aquifer recharge from irrigation in a semiarid agricultural valley in New Mexico. *Journal of Hydrologic Engineering* **18**, 1219–1230 (2013).
 39. Wakode, H.B., Baier, K., Jha, R. & Azzam, R. Impact of urbanization on groundwater recharge and urban water balance for the city of Hyderabad, India. *International Soil and Water Conservation Research* **6**, 51–62 (2018).
 40. Meredith, E. & Blais, N. Quantifying irrigation recharge sources using groundwater modeling. *Agricultural Water Management* **214**, 9–16 (2019).
 41. Dillon, P., Stuyfzand, P., Grischek, T., Lloria, M., Pyne, R.D.G., Jain, R.C., Bear, J., Schwarz, J., Wang, W., Fernandez, E. & Stefan, C. Sixty years of global progress in managed aquifer recharge. *Hydrogeology Journal* **27**, 1–30 (2019).
 42. Scanlon, B.R., Reedy, R.C., Faunt, C.C., Pool, D. & Uhlman, K. Enhancing drought resilience with conjunctive use and managed aquifer recharge in California and Arizona. *Environmental Research Letters* **11**, 035013 (2016).

43. Stefan, C. & Ansems, N. Web-based global inventory of managed aquifer recharge applications. *Sustainable Water Resources Management* **4**, 153–162 (2018).
44. Koh, E.H., Kaown, D., Mayer, B., Kang, B.R., Moon, H.S. & Lee, K.K. Hydrogeochemistry and isotopic tracing of nitrate contamination of two aquifer systems on Jeju Island, Korea. *Journal of Environmental Quality* **41**, 1835–1845 (2012).
45. Han, D., Currell, M.J. & Cao, G. Deep challenges for China's war on water pollution. *Environmental Pollution* **218**, 1222–1233 (2016).
46. Shuler, C.K., Dulai, H., DeWees, R., Kirs, M., Glenn, C.R. & El-Kadi, A. I. Isotopes, microbes, and turbidity: A multi-tracer approach to understanding recharge dynamics and groundwater contamination in a Basaltic Island Aquifer. *Groundwater Monitoring & Remediation* **39**, 20–35 (2019).
47. Berg, M., Trang, P.T.K., Stengel, C., Buschmann, J., Viet, P.H., Van Dan, N., Giger, W. & Stüben, D. Hydrological and sedimentary controls leading to arsenic contamination of groundwater in the Hanoi area, Vietnam: The impact of iron-arsenic ratios, peat, river bank deposits, and excessive groundwater abstraction. *Chemical Geology* **249**, 91–112 (2008).
48. Jasechko, S. Partitioning young and old groundwater with geochemical tracers. *Chemical Geology* **427**, 35–42 (2016).
49. Jasechko, S., Perrone, D., Befus, K.M., Cardenas, M.B., Ferguson, G., Gleeson, T., Luijendijk, E., McDonnell, J.J., Taylor, R.G., Wada, Y. & Kirchner, J.W. Global aquifers dominated by fossil groundwaters but wells vulnerable to modern contamination. *Nature Geoscience* **10**, 425–429 (2017).
50. Jasechko, S. Global isotope hydrogeology—Review. *Reviews of Geophysics* **57**, 835–965 (2019).
51. Lindholm, G. F. Summary of the Snake River Plain regional aquifer-system analysis in Idaho and eastern Oregon. U.S Geological Survey Professional Paper 1408-A. 59 pp. Accessed March 1, 2021 via <https://pubs.usgs.gov/pp/1408a/report.pdf> (1996).
52. Driscoll, D. G., Carter, J. M., Williamson, J. E. & Putnam, L. D. Hydrology of the Black Hills area, South Dakota. US Geological Survey Water-Resources Investigations Report 2002-4094, 158 pp. Accessed February 16, 2021 from <https://pubs.usgs.gov/wri/wri024094/pdf/wri024094.pdf> (2002).
53. Long, A.J., Thamke, J.N., Davis, K.W. & Bartos, T.T. Groundwater availability of the Williston Basin, United States and Canada. US Geological Survey Professional Paper 1841, 42 p., Accessed February 16, 2021 from <https://doi.org/10.3133/pp1841> (2018).
54. Prior, J.C., Boekhoff, J.L., Howes, M.R., Libra, R.D. & VanDorpe, P.E. Iowa's groundwater basics. Iowa Department of Natural Resources Report, 92 pp. Accessed November 29, 2021 from https://s-iihr34.iihr.uiowa.edu/publications/uploads/2014-08-24_08-08-21_es-06.pdf (2003).
55. Westjohn, D.B. & Weaver, T.L. Hydrogeologic framework of the Michigan Basin regional aquifer system. US Geological Survey Professional Paper 1418, 55 pp. Accessed November 29, 2021 from <https://pubs.usgs.gov/pp/1418/report.pdf> (1998).
56. Smolensky, D.A., Buxton, H.T. & Shernoff, P.K. Hydrologic framework of Long Island, New York. U.S. Geological Survey Hydrologic Atlas 709, 3 plates, Accessed June 15, 2022 from <https://pubs.usgs.gov/ha/709/plate-1.pdf> (1990).
57. Winner Jr., M.D. & Coble, R.W. Hydrogeologic framework of the North Carolina Coastal Plain aquifer system. US Geological Survey Report 87-690, 167 pp. Accessed April 1, 2021 from <https://pubs.usgs.gov/of/1987/0690/report.pdf> (1989).

58. Aucott, W.R. Hydrology of the Southeastern Coastal Plain Aquifer System in South Carolina and Parts of Georgia and North Carolina. U.S Geological Survey Professional Paper 1410-E, 95 pp. Accessed March 31, 2021 from <https://pubs.usgs.gov/pp/1410e/report.pdf> (1996).
59. Mashburn, S.L., Ryter, D.W., Neel, C.R., Smith, S.J. & Correll, J.S. Hydrogeology and simulation of ground-water flow in the Central Oklahoma (Garber-Wellington) Aquifer, Oklahoma, 1987 to 2009, and simulation of available water in storage, 2010–2059. US Geological Survey Scientific Investigations Report 2013–5219, 92 pp. Accessed May 14, 2021 from https://pubs.usgs.gov/sir/2013/5219/pdf/sir20135219_v2.0.pdf (2014).
60. Blandford, T. N., Kuchanur, M., Standen, A., Ruggiero, R., Calhoun, K. C., Kirby, P. & Shah, G. Groundwater Availability Model of the Edwards-Trinity (High Plains) Aquifer in Texas and New Mexico. Texas Water Development Board. Accessed June 20, 2022 from <https://www.twdb.texas.gov/groundwater/aquifer/minors/edwards-trinity-high-plains.asp> (2008).
61. Robertson, A.J., Matherne, A.M., Pepin, J.D., Ritchie, A.B., Sweetkind, D.S., Teeple, A.P., Granados-Olivas, A., García-Vásquez, A.C., Carroll, K.C., Fuchs, E.H. & Galanter, A.E. Mesilla/Conejos-Médanos Basin: US-Mexico Transboundary Water Resources. *Water* 14, 134 (2022).
62. Hanson, R.T., Martin, P. & Koczot, K.M. Simulation of ground-water/surface-water flow in the Santa Clara-Calleguas ground-water basin, Ventura County, California. Water-Resources Investigations Report 2002-4136, 172 pp. Accessed March 20, 2021 from <https://pubs.usgs.gov/wri/wri024136/wri024136.pdf> (2002).
63. Page, R. W. & Balding, G. O. Geology and quality of water in the Modesto-Merced area, San Joaquin Valley, California, with a brief section on hydrology. US Geological Survey Water-Resources Investigations Report 73-6, 77 pp. Accessed June 16, 2022 via <https://pubs.er.usgs.gov/publication/wri736> (1973).
64. Manning, A.H. & Solomon, D.K. An integrated environmental tracer approach to characterizing groundwater circulation in a mountain block. *Water Resources Research* 41, W12412, doi:10.1029/2005WR004178 (2005).
65. Carlson, M. A., Lohse, K. A., McIntosh, J. C. & McLain, J. E. Impacts of urbanization on groundwater quality and recharge in a semi-arid alluvial basin. *Journal of Hydrology* 409, 196–211 (2011).
66. Hamlin, S.N., Belitz, K. & Johnson, T. Occurrence and Distribution of Volatile Organic Compounds and Pesticides in Ground Water in Relation to Hydrogeologic Characteristics and Land Use in the Santa Ana Basin, Southern California. US Geological Survey Scientific Investigations Report 2005-5032, 49 pp. Accessed May 31, 2022 via <https://pubs.usgs.gov/sir/2005/5032/sir2005-5032.pdf> (2005).
67. Bennett, G.L., Fram, M.S., Belitz, K. & Jurgens, B.C. Status and Understanding of Groundwater Quality in the Northern San Joaquin Basin, 2005: California GAMA Priority Basin Project. US Geological Survey Scientific Investigations Report 2010–5175, 96 pp. Accessed May 31, 2022 via <https://pubs.usgs.gov/sir/2010/5175/pdf/sir20105175.pdf> (2010).
68. Cederberg, J.R., Gardner, P.M. & Thiros, S.A. Hydrology of Northern Utah Valley, Utah County, Utah, 1975–2005. US Geological Survey Scientific Investigations Report 2008–5197, 128 pp. Accessed May 31, 2022 via <https://pubs.usgs.gov/sir/2008/5197/pdf/sir2008-5197.pdf> (2008).
69. Hansen, J.A., Jurgens, B.C. & Fram, M.S. Quantifying anthropogenic contributions to century-scale groundwater salinity changes, San Joaquin Valley, California, USA. *Science of the Total Environment* 642, 125–136 (2018).

70. Thiros, S.A. & Manning, A.H. Quality and sources of ground water used for public supply in Salt Lake Valley, Salt Lake County, Utah, 2001. US Geological Survey Water-Resources Investigations Report 2003-4325, 24 pp. Accessed May 31, 2022 via https://pubs.usgs.gov/pp/1781/pdf/pp1781_section2.pdf (2003).
71. Landon, M. K., Jurgens, B.C., Katz, B.G., Eberts, S.M., Burow, K.R. & Crandall, C.A. Depth-dependent sampling to identify short-circuit pathways to public-supply wells in multiple aquifer settings in the United States. *Hydrogeology Journal* **18**, 577–593 (2010).
72. Ferguson, G., Cuthbert, M. O., Befus, K., Gleeson, T., & McIntosh, J. C. (2020). Rethinking groundwater age. *Nature Geoscience* **13**, 592-594.
73. Jurgens, B.C., Faulkner, K., McMahon, P.B., Hunt, A. G., Casile, G., Young, M.B., Belitz, K. (2022). Over a third of groundwater in USA public-supply aquifers is Anthropocene-age and susceptible to surface contamination. *Communications Earth & Environment*, **3**, 1-9.

Acknowledgments. This material is based upon work supported by the National Science Foundation under Grant No. EAR-2048227- (S.J.). This research was supported by funding from the Zegar Family Foundation- (S.J.). This material is based upon work ~~was~~ supported by ~~funding from~~ the U.S. Geological Survey (USGS) through the California Institute for Water Resources (CIWR) ~~Water Research Program under Grant/Cooperative Agreement No. G21AP10611-00 (S.J.).~~ The views and conclusions contained in this document are those of the authors and should not be interpreted as representing the opinions or policies of USGS/CIWR. Mention of trade names or commercial products does not constitute their endorsement by USGS/CIWR. The authors acknowledge the Jack and Laura Dangermond Preserve, (<https://doi.org/10.25497/D7159W>), the Point Conception Institute, and the Nature Conservancy for their support of this research.

Formatted: Font: Times New Roman, English (United Kingdom)

Ethics declarations. The authors declare no competing interests.

Formatted: Font: Not Bold

Author information – Affiliations. ^a Bren School of Environmental Science and Management, University of California, Santa Barbara, 93106, California, USA ^b Department of Geology, University of Puerto Rico, Mayagüez, 00682, Puerto Rico — ^a *Melissa Thaw*, ^a *Merhawi GebreEgziabher*, ^{a,b} *Jobel Y. Villafañe-Pagán*, ^a *Scott Jasechko*

Author Contributions Statement. M.T. and S.J. contributed equally to the manuscript. M.T. and S.J. co-generated hypotheses, co-developed methods, and co-wrote the manuscript. M.T. and S.J. calculated the depth below which modern groundwater is scarce. M.G. digitized hydrogeologic cross sections and completed the literature review to categorize geologic formations (e.g., as aquifers or as aquitards). J.Y. V-P. georeferenced maps from reviewed literature to approximate the location of each hydrogeologic cross section. S.J. estimated the depth to confined conditions for each study area.

Corresponding author. Correspondence to Scott Jasechko (jasechko@ucsb.edu)

Figure Captions

Fig. 1. Well water tritium (^3H) measurements across the contiguous United States. a) Light yellow shades represent lower tritium activities (below 1 tritium unit), orange shades represent mid-range tritium activities (1 to 10 tritium units) and red shades represent well waters with high tritium activities (exceeding 10 tritium units; 1 tritium unit is ~ 0.118 Bq/L). Light grey polygons underlying the yellow-red points are aquifer system boundaries published by GebreEgziabher et al. (ref.¹⁹; data available via <https://www.hydroshare.org/resource/d2260651b51044d0b5cb2d293d21af08/>). Panels (b-o) display hydrostratigraphy via cross sections for $n=14$ of the $n=74$ aquifer systems that we studied. The presented cross sections are based on figures and lithologic descriptions presented for the b) Boise Valley and Homedale Area within the broader Snake River Plain in ref.⁵¹, c) Black Hills Uplift in ref.⁵², d) Williston Basin in ref.⁵³, e) Eastern Dakota Aquifer System in ref.⁵⁴, f) Michigan Basin in ref.⁵⁵, g) Long Island in ref.⁵⁶, h) Castle Hayne Aquifer System in ref.⁵⁷, i) Lower Coastal Plain subarea of the broader Floridan Aquifer System in ref.⁵⁸, j) Garber-Wellington Aquifer System in ref.⁵⁹, k) Southern High Plains in ref.⁶⁰, l) Mesilla Valley in ref.⁶¹, m) Santa Clara-Calleguas Basin in ref.⁶², n) Tulare Basin subarea of the broader California Central Valley Aquifer System in ref.⁶³, and the o) Salt Lake Valley in ref.⁶⁴. Cross sections for each of our $n=74$ study aquifer systems are presented in Supplementary Figs. 99-240 (locations of each of the $n=74$ hydrogeologic cross sections we examined are displayed in Supplementary Note 8).

Fig. 2. Spatial patterns of the depth below which modern groundwater is scarce among US aquifer systems. Yellow-blue shades represent the depth below which 60% (panel a), 70% (panel b) or 80% (panel c) of samples contain minimal modern groundwater (defined here as well water samples containing less than 25% modern groundwater). **a)** The median depth below which more than 60% of samples contain minimal modern groundwater among n=74 studied aquifer systems (with sufficient groundwater ^3H data) is 38 meters, and the lower-upper quartile range is 12-92 meters. **b)** The median depth below which more than 70% of samples contain minimal modern groundwater among n=68 studied aquifer systems is 60 meters, and the lower-upper quartile range is 25-113 meters. **c)** The median depth below which more than 80% of samples contain minimal modern groundwater among n=61 studied aquifer systems is 75 meters, and the lower-upper quartile range is 49-149 meters. For a map with labels identifying aquifer systems see Supplementary Fig. 98.

Fig. 3. The depth below which modern groundwater is scarce tends to be deeper where annual groundwater withdrawals are high. a) Relationship between estimated annual groundwater withdrawals²¹ and the depth below which modern groundwater is scarce. Each point represents one aquifer system. Annual groundwater withdrawals were normalized by aquifer area, such that the units (mm/year) can be interpreted as all withdrawn groundwater during the year 2015 expressed as saturated layer if it were spread evenly across the study area (for study area boundaries see panel b). The Spearman rank correlation coefficient (ρ) of the data presented in panel a is $\rho = 0.42$ (Spearman P-value < 0.001). The y-axis values corresponding to each coloured point are the depth below which >70% of samples have minimal (less than 25%) modern groundwater; vertical error bars extend to the depth below which >60% of samples have minimal (less than 25%) modern groundwater (shallower depth—i.e., top of grey error bar as displayed in plot) and the depth below which >80% of samples have minimal (less than 25%) modern groundwater (deeper depth—i.e., bottom of grey error bar as displayed in plot; see legend in upper-left corner of plot). We only plot the $n=74$ points representing the aquifer systems for which we had sufficient data to determine the depth below which >70% of samples have minimal (less than 25%) modern groundwater. Points are colour coded by the estimated depth to confined conditions (see legend in panel b). **b)** Estimated depth to confined conditions in each of our $n=74$ study aquifer systems. Each polygon on the map represents one study area. Light blue colors represent shallower depths to confined conditions; darker blue shades represent deeper depths to confined conditions. Depths to confined conditions were estimated on the basis of up to three data sources: (i) USGS-defined well conditions (i.e., wells defined as tapping unconfined versus confined conditions by the US Geological Survey), (ii) digitization and evaluation of hydrogeologic cross sections derived from local-scale reports, and (iii) quotes within local-scale reports pertaining to the prevalence of confined conditions (for details see Supplementary Note 3).

Fig. 4. Schematic of some of the processes that may influence vertical distributions of modern groundwater. Artificial groundwater recharge (left side of schematic) can increase the magnitude of downward-oriented vertical hydraulic gradients and potentially drive modern groundwater to deeper depths. Artificial recharge can derive from urban waters (e.g., Tucson Basin⁶⁵), spreading basins associated with managed aquifer recharge projects (e.g., Upper Santa Ana Basin⁶⁶), excess irrigation waters (e.g., San Joaquin Basin⁶⁷), or leaky surface water conveyance infrastructure such as canals (e.g., Utah Lake Valley⁶⁸). The construction of groundwater wells and their use via pumping (right side of schematic) may help modern waters enter deeper wells. For example, deep wells that have long perforated intervals may draw both shallow (disproportionately modern) and deep (disproportionately pre-modern) groundwater (e.g., Tulare Basin⁶⁹). Further, pumping from wells can alter vertical hydraulic gradients, potentially drawing shallow modern groundwater to deeper depths (e.g., Salt Lake Valley⁷⁰). The modern groundwater may move downward to deeper depths in the aquifer system via natural pore spaces (e.g., discontinuities in aquitards; e.g., Eastern Mississippi Embayment³⁶) or via conduits created by constructed wells themselves (e.g., Northern High Plains⁷¹). For further discussion and schematics of modern groundwater distributions see refs.^{50,72,73}. For a review of some of the studies that have posited one or more of these mechanisms as potential explanations for the distribution of modern groundwater see Supplementary Note 6. For a more expansive version of this figure see Supplementary Fig. 243.

Ethics declarations. The authors declare no competing interests.

Data Availability. Aquifer system boundaries from ref.⁴⁹ are available to download via CUAHSI HydroShare: <https://www.hydroshare.org/resource/d2260651b51044d0b5eb2d293d21af08/>. Groundwater withdrawal estimates (county scale) are available via <https://pubs.er.usgs.gov/publication/cir1441>. Well water tritium data are available via <https://www.waterqualitydata.us/portal>.

Formatted: Font: Not Bold

~~**Code availability.** The analyses presented here do not depend on specific code. Our approach can be reproduced following the procedures described in the Methods.~~